# Dimension-adapted Momentum Outscales SGD

**Damien Ferbach**[*]
Mila & Université de Montréal
ferbach.damien@gmail.com

**Katie Everett**
Google DeepMind & MIT
everettk@google.com

**Gauthier Gidel**[†]
Mila & Université de Montréal
gidelgau@mila.quebec

**Elliot Paquette**[‡]
McGill University
elliot.paquette@mcgill.ca

**Courtney Paquette**[‡]
Google DeepMind & McGill University
courtney.paquette@mcgill.ca

## Abstract

We investigate scaling laws for stochastic momentum algorithms with small batch on the power law random features model, parameterized by data complexity, target complexity, and model size. When trained with a stochastic momentum algorithm, our analysis reveals four distinct loss curve shapes determined by varying data-target complexities. While traditional stochastic gradient descent with momentum (SGD-M) yields identical scaling law exponents to SGD, dimension-adapted Nesterov acceleration (DANA) improves these exponents by scaling momentum hyperparameters based on model size and data complexity. This *outscaling* phenomenon, which also improves compute-optimal scaling behavior, is achieved by DANA across a broad range of data and target complexities, while traditional methods fall short. Extensive experiments on high-dimensional synthetic quadratics validate our theoretical predictions and large-scale text experiments with LSTMs show DANA's improved loss exponents over SGD hold in a practical setting.

## 1 Introduction

When pretraining large neural networks, the loss typically scales like a power law with respect to the amount of data, number of parameters, and total amount of compute [54]. Scaling laws for the loss function $\mathscr{P}(\theta)$ in their simplest form are $\mathscr{P}(\theta_t) \asymp t^{-\sigma} + d^{-\tau}$ where $\{\theta_t \in \mathbb{R}^d\}$ is a sequence of iterates generated by a stochastic algorithm.[4] The exponents $\sigma$ and $\tau$ are of practical importance because they control the number of samples and parameters needed to attain a desired loss value.

While model architectures and training methods have advanced rapidly, it has long been unclear whether innovations in *optimization algorithms* could fundamentally change the exponents of these power laws [48]. Some evidence suggests that major advances like the Adam optimizer [56] primarily improve the constants in the scaling law rather than improving its exponent [48].

Moreover, recent work has extensively investigated how various algorithmic parameters such as learning rates [103] and batch sizes [70] should scale as the model sizes and compute grow. However, momentum parameters are typically treated as fixed constants [93] rather than dimension/compute-dependent quantities, despite their widespread use in large model training. This leads to the question:

---

[*]Corresponding author; website: https://damienferbach.github.io/.

[†]Canada CIFAR AI Chair

[‡]The authors contributed equally to the paper.

[4]Notation: $f \asymp g$ means there exist constants $c$ and $C$ (both independent of $d$) such that $cg \leq f \leq Cg$. We use $(x)_+$ to denote $\max\{0, x\}$.

39th Conference on Neural Information Processing Systems (NeurIPS 2025).

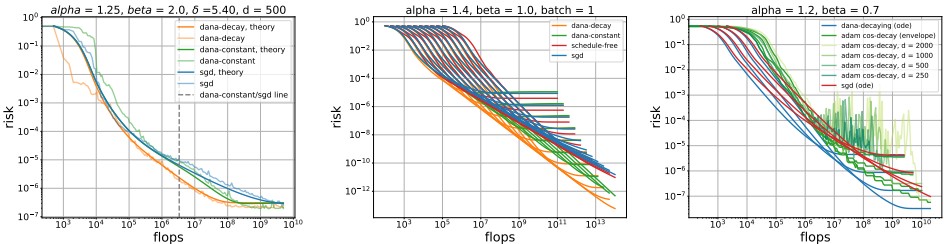

Figure 1: **For empirical runs and deterministic ODE (7) simulations of PLRF, DANA outscales SGD while Schedule-Free SGD and Adam do not.** *(left)* SGD & DANA fixed $d$, single run. Gray dashed line indicates transition ($t \asymp 1/\gamma_3 \asymp d$) where DANA-constant($\kappa_2 = 1$) shifts from SGD-like behavior to acceleration. Deterministic ODE predictions (bold curves) match single runs of the stochastic algorithms (faded curves). *(middle)* Deterministic ODEs for SGD, DANA and Schedule-free SGD [33], across $d = \{100 \times 2^i\}, i = 1, .., 10$. DANA-decaying ($\kappa_3 = \frac{1}{2\alpha}$) outscales DANA-constant, which itself outscales SGD. Schedule-free SGD scales similarly to SGD. *(right)* ODEs show Adam cosine decay [56] appears to have the same scaling law as SGD. See Sec. M for details. In all figures, batch size is 1.

*Can one select hyperparameters of stochastic momentum algorithms as a function of model size and data to provably change the exponents in the scaling laws for the loss?*

In this work, we answer in the affirmative, mathematically showing that one can indeed improve the scaling law exponents over standard stochastic gradient descent (SGD) on a simple power law random features (PLRF) model.

**Outscaling.** To address the question of scaling, let us consider the following learning problem,

$$\min_{\theta \in \mathbb{R}^d} \left\{ \mathscr{P}(\theta) = \mathbb{E}_x[\mathscr{R}(\theta; x)] \right\}, \text{ where } \mathscr{R} : \mathbb{R}^d \to \mathbb{R}, \tag{1}$$

where $d$ is the parameter count, and where $x$ is drawn from an unknown distribution.

A *training regime*, $t \asymp d^\ell, \ell > 0$, is a scaling of iterations (or samples) to parameters. There are many examples of training regimes, e.g., the *proportional regime* ($t \asymp d$) or the *compute-optimal regime*, in which one selects the $\ell$ that yields the best loss under a fixed compute budget (see Sec. 5). Now suppose the loss under an algorithm follows the scaling law

$$\mathscr{P}(t, d) \overset{\text{def}}{=} \mathscr{P}(\theta_t, d) \asymp t^{-\sigma} + d^{-\tau} \quad \text{and suppose } t \asymp d^\ell, \ell > 0 \text{ is a training regime.} \tag{2}$$

Then under this training regime, the loss satisfies $\mathscr{P}(d^\ell, d) \asymp d^{-\min\{\ell\sigma, \tau\}}$. We call the absolute exponent on $d$, the *loss exponent*. For a given training regime, we say an algorithm *outscales* another algorithm if the loss exponent is larger.

We emphasize that this notion of outscaling differs in one key aspect from the more traditional notion of "acceleration" from optimization theory. Acceleration is typically formulated for a fixed-dimensional problem with constants that can have large $d$-dependence (e.g., $\|\theta_0 - \theta^\star\|^2$). In other words, acceleration generally denotes outperformance when $t \to \infty$ and $d = O(1)$.

In this work, we are interested in scaling laws of one-pass, mini-batch SGD with momentum with batch size $B$. At iteration $t \geq 0$, we generate independent samples $\{x_{t+1}^i\}_{i=1}^B$ and update:

$$y_t = (1 - \Delta(t))y_{t-1} + \gamma_1(t; d) \sum_{i=1}^B \nabla \mathscr{R}(\theta_t; x_{t+1}^i),$$
$$\theta_{t+1} = \theta_t - \gamma_2(t; d) \sum_{i=1}^B \nabla \mathscr{R}(\theta_t; x_{t+1}^i) - \gamma_3(t; d)y_t, \tag{Gen-Mom-SGD}$$

where $\Delta(t) : [0, \infty) \to [0, \infty)$ is a momentum hyperparameter and $\gamma_i(t; d) : [0, \infty) \to [0, \infty)$ are learning rates. This framework incorporates classical SGD and SGD-Momentum by setting

$$\gamma_1(t) \equiv 1, \quad \gamma_2(t; d) \equiv \gamma_2 = \tilde{\gamma}_2 d^{-\kappa_1}, \quad \gamma_3(t; d) \equiv \gamma_3 = \tilde{\gamma}_3 d^{-\kappa_2}, \quad \Delta(t) \equiv \delta, \quad \kappa_i \geq 0 \quad \text{(SGD-M)}$$

as well as stochastic Nesterov, Schedule-Free SGD [33], and accelerated SGD [52, 94]. For a detailed discussion on related work see Section A and Tables 2 & 3.

**Main Contributions.** In this work, we analyze Gen-Mom-SGD under the power law random features (PLRF) – a four-parameter model with data complexity $(1/\alpha)$, target complexity $(1/\beta)$,[5]

---

[5]See Assumption 2 for formal definition of $\alpha$ and $\beta$ in the context of the power law random features model.

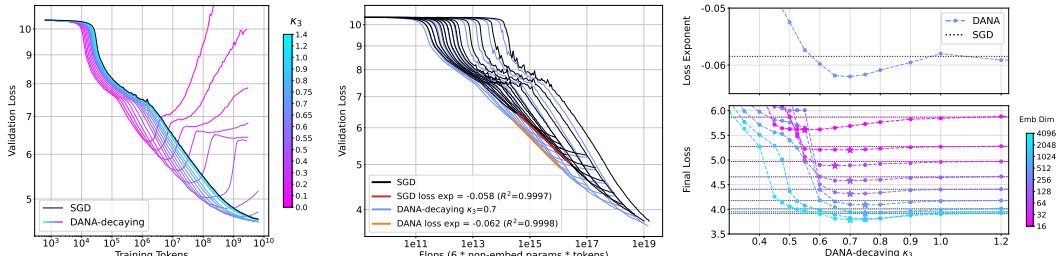

Figure 2: **DANA-decaying improves the loss exponent on LSTM language models.** *(left)* Sweeping DANA-decaying $\kappa_3$ shows stability and divergence similar to PLRF (Fig. 3a). *(center)* DANA $\kappa_3 = 0.7$ maximizes the compute-optimal loss exponent and outscales SGD. *(right)* Compute-optimal loss exponents *(top)* and validation loss for final iterate *(bottom)* vs DANA $\kappa_3$. All loss exponents (Fig. 22) have $R^2 \geq 0.984$ and vary smoothly across $\kappa_3$, traversing the divergent, outscaling, and SGD-like regimes seen in PLRF (Fig. 6 x-axis). Schedule-Free SGD is $\kappa_3 = 1.0$ and matches SGD loss exponent. SGD-M matches SGD (Fig. 24). See Sec. L.

model parameter count $d$, and hidden dimensionality $v$ – extensively studied for its scaling properties [11, 69]. We derive a scaling rule for hyperparameters of Gen-Mom-SGD that improves loss exponents over SGD across much of the $(\alpha, \beta)$-phase plane under a variety of training regimes. We denote this scaling hyperparameter rule as *dimension-adapted Nesterov acceleration (DANA)*:

$$\gamma_1(t;d) \equiv 1, \ \gamma_2(t;d) = \tilde{\gamma}_2 d^{-\kappa_1}, \ \gamma_3(t;d) = \tilde{\gamma}_3 d^{-\kappa_2}(1+t)^{-\kappa_3}, \ \Delta(t) = \delta(1+t)^{-1}, \quad \text{(DANA)}$$

where $\kappa_i \geq 0$. These hyperparameters will further be made explicit for the PLRF.

To show this improvement in the exponents, we show that the loss curves for PLRF can be described exactly by a system of differential equations, and derive precise theoretical scaling laws for SGD-M and DANA. Using dimension- and data-dependent hyperparameters, DANA outscales SGD (see Fig. 1) above the high-dimensional line ($2\alpha > 1$) while performing no worse than SGD elsewhere. In contrast, traditional SGD-M, with fixed, non-scaling momentum, $\Delta$, produces identical scaling laws to standard SGD across all regimes (see Fig. 24). DANA-constant ($\kappa_3 = 0$) employs a decaying momentum schedule $(1+t)^{-1}$ with the momentum learning rate given as $\gamma_3 \asymp \gamma_2 \cdot 1/d$, and outscales SGD for $2\alpha > 1$ under many regimes. DANA-decaying ($\kappa_2 = \kappa_1 = 0, \kappa_3 > 0$) replaces $d$ with a time-dependent *effective dimension*, resulting in a schedule for $\gamma_3$ that outscales SGD for all regimes and all $2\alpha > 1$, succeeding where DANA-constant cannot.

We then investigate the compute-optimal regime, explicitly deriving compute-optimal parameter, loss, and data exponents. While the empirical Chinchilla laws [50] suggest compute-optimality occurs at $d \asymp t$, we show DANA-decaying (for $2\alpha > 1$) has a *different* compute-optimal relationship between $d$ and $t$, emphasizing that outscaling can occur outside the Chinchilla regime. Moreover, for some $(\alpha, \beta)$, DANA reduces the data exponent needed to reach compute-optimality compared to SGD.

We perform extensive PLRF experiments in Sec. K to validate our theory, showing excellent numerical agreement for loss curves and compute-optimal exponents with the analyzed ODEs. Finally, we train LSTMs on text data (Fig. 2) showing the DANA loss exponents (Fig. 2c & 22) vary smoothly over $\kappa_3$ and recover the divergent, outscaling, and SGD-like regimes predicted theoretically by Fig. 6.

While we show the existence of stochastic algorithms that provably outscale for $2\alpha > 1$, it remains an open question as to whether outscaling SGD can occur in the high-dimensional setting ($2\alpha < 1$).

## 2 The power law random features model (PLRF)

In this work, we analyze a four-parameter model called *power law random features* (PLRF) (3) [11, 69, 80], which exhibits rich behavior and phenomenologically captures many aspects of scaling law setups [16, 63, 80]. For a data vector $x \in \mathbb{R}^v$ we embed this vector in $\mathbb{R}^d$ using a matrix $W \in \mathbb{R}^{v \times d}$ and construct noiseless targets by dotting a fixed $b \in \mathbb{R}^v$ with the sample $x$. This leads to the formal problem statement:

$$\min_{\theta \in \mathbb{R}^d} \left\{ \tfrac{1}{2} \mathscr{P}(\theta) \stackrel{\text{def}}{=} \tfrac{1}{2} \mathbb{E}_x \left[ (\langle W^T x, \theta \rangle - \langle x, b \rangle)^2 \right] \right\}. \tag{3}$$

The matrix $W$ allows the model to have variable capacity ($d$) independent of the data dimension, and we choose the matrix $W$ to have entries distributed as $N(0, 1/d)$. The key structural assumptions are:

**Assumption 1** (Data and targets, $\alpha$ and $\beta$)**.** *The samples $x \in \mathbb{R}^v$ are distributed according to $(x_j) \sim j^{-\alpha} z_j$ for all $1 \le j \le v$ and $\{z_j\}_{j=1}^v \sim N(0, 1)$. The targets are scalars constructed by dotting the sample $x$ with a signal $b \in \mathbb{R}^v$ whose entries $(b_j) = j^{-\beta}$.*

Power law type data distributions are ubiquitous in language, vision, and many other tasks, and these distributions are largely responsible for making this model phenomenologically similar to scaling law setups [69, Fig.2,3]. Without the random matrix $W$, $(\alpha, \beta)$ are related to what is known in the literature as source and capacity conditions [22, 23, 36, 81] (see Sec. A and Tab. 1 for details).

The hidden dimensionality $v$ is assumed to be large and proportionate to $d$, so that $v/d \to r \in (1, \infty)$. In the case that $2\alpha > 1$, this assumption can be relaxed, in that one can take $v$ larger as a function of $d$ or even $v = \infty$. It should be noted that in many scaling law setups, such as [50], the task scales with the parameter count, so that it is natural to assume $v$ grows as $d$ grows.[6]

## 3 Continuized analysis of general stochastic momentum algorithms

Continuized frameworks are widely used for the analysis of momentum algorithms (see especially [38, 78, 92, 99]). When run on the PLRF model, the algorithm class (Gen-Mom-SGD) has a loss curve that can be described exactly by a system of differential equations; Section C.1 and (22) for details and derivation.

We use a common probabilistic trick for continuizing a discrete process called *Poissonization* in which we let $(N_t : t \ge 0)$ be a standard Poisson process, and then define $Y_t = y_{N_t}$ and $\Theta_t = \theta_{N_t}$. Now we let $(\lambda_j, \omega_j)_{j=1}^d$ be the eigenvalue-eigenvector pairs for $\check{K} \in \mathbb{R}^{d \times d}$, which is the covariance of the projected data $W^\top x$, and let $\Theta^*$ be the minimizer of $\mathscr{P}(\Theta)$. We then introduce the system of variables, with the expectations taken over all randomness except $W$,

$$\rho_j^2(t) \stackrel{\text{def}}{=} \mathbb{E}\left[\langle \omega_j, \Theta_t - \Theta^* \rangle^2\right], \ \xi_j^2(t) \stackrel{\text{def}}{=} \mathbb{E}\left[\langle \omega_j, Y_t \rangle^2\right], \text{ and } \chi_j(t) \stackrel{\text{def}}{=} \mathbb{E}\left[\langle \omega_j, \Theta_t - \Theta^* \rangle \langle \omega_j, Y_t \rangle\right]. \quad (4)$$

In terms of these variables, we recover the expected loss by summing over the components of $\rho_j^2(t)$.

$$\mathcal{P}(t) \stackrel{\text{def}}{=} \mathbb{E}\left[\mathscr{P}(\Theta_t)\right] = \mathbb{E}\left[\mathscr{P}(\Theta^*)\right] + \sum_{j=1}^d \lambda_j \rho_j^2(t). \quad (5)$$

Then we can derive a coupled linear system of differential equations:

$$\nu(t; \lambda_j) \stackrel{\text{def}}{=} (\rho_j^2(t), \xi_j^2(t), \chi_j(t))^\top, \quad \frac{\mathrm{d}}{\mathrm{d}t} \nu(t; \lambda_j) = \Omega(t; \lambda_j) \times \nu(t; \lambda_j) + \mathcal{P}(t) g(t; \lambda_j), \quad (6)$$

for an explicit matrix $\Omega$ and vector $g$ which are polynomials in $(\lambda_j, \Delta, B, \gamma_1, \gamma_2, \gamma_3)$ (see (22) for the full formula).

These equations are sufficiently complicated that we simplify the ODE before performing the analysis. This can be viewed as taking a "high-dimensional limit" (see Remark C.3), or, in short, as dropping all terms from $\Omega$ and $g$ which are more than first order in $(\Delta, \lambda_j, \gamma_3)$ to produce:

$$\Omega(t; \lambda_j) \stackrel{\text{def}}{=} \begin{pmatrix} -2\gamma_2(t) B \lambda_j & 0 & -2\gamma_3(t) \\ 0 & -2\Delta(t) & 2\gamma_1(t) B \lambda_j \\ \gamma_1(t) B \lambda_j & -\gamma_3(t) & -\Delta(t) - \gamma_2(t) B \lambda_j \end{pmatrix} \quad \text{and} \quad g(t; \lambda_j) \stackrel{\text{def}}{=} \lambda_j B \begin{pmatrix} \gamma_2^2(t) \\ \gamma_1^2(t) \\ 0 \end{pmatrix}.$$

We formulate all results going forward for the simplified ODEs; see Sec. K and Fig. 8 for extensive numerical validation of the simplified ODEs as a model for the risk curves of Gen-Mom-SGD. We also note that these simplified ODEs correspond exactly to the risk evolution under the SDE model of Gen-Mom-SGD:

$$\begin{aligned} \mathrm{d}Y_t &= -\Delta(t) Y_t \, \mathrm{d}t + \gamma_1(t) \left( \tfrac{B}{2} \nabla \mathscr{P}(\Theta_t) \, \mathrm{d}t + \sqrt{B \check{K} \mathscr{P}(\Theta_t)} \, \mathrm{d}\mathcal{B}_t^{(1)} \right), \\ \mathrm{d}\Theta_t &= -\gamma_2(t) \left( \tfrac{B}{2} \nabla \mathscr{P}(\Theta_t) \, \mathrm{d}t + \sqrt{B \check{K} \mathscr{P}(\Theta_t)} \, \mathrm{d}\mathcal{B}_t^{(2)} \right) - \gamma_3(t) Y_t \, \mathrm{d}t. \end{aligned} \quad (7)$$

---

[6]In the case $2\alpha < 1$, there is a larger range of scalings of $v$ that are possible, with $d \ll v \ll d^{1/(1-2\alpha)}$.

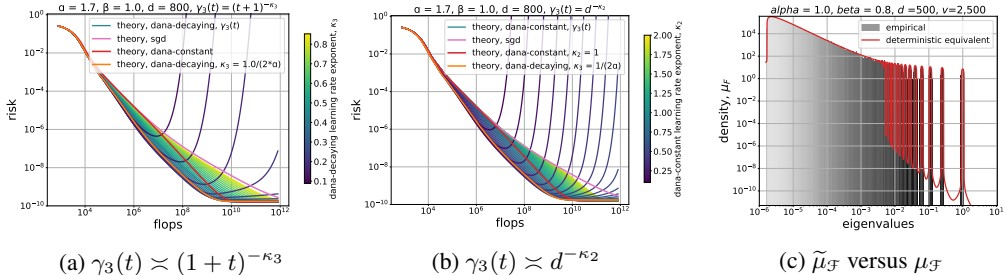

(a) $\gamma_3(t) \asymp (1+t)^{-\kappa_3}$      (b) $\gamma_3(t) \asymp d^{-\kappa_2}$      (c) $\widetilde{\mu}_{\mathcal{F}}$ versus $\mu_{\mathcal{F}}$

Figure 3: **(left) & (center) DANA sweeps of** $\kappa_2, \kappa_3$**.** A $\kappa$ bigger than (stability) $\Rightarrow$ divergence after some $t$; DANA-decaying w/ $\kappa_3 = 1/(2\alpha)$ is the envelope of the divergent algorithms. $\kappa_3 = 1/(2\alpha)$, $\kappa_2 = 0$ is optimal in DANA. **(right) Deterministic equivalent** $\mu_{\mathcal{F}}$ (8) vs empirical density estimate match. See Sec. M for details.

Here $\mathrm{d}\mathcal{B}_t^{(1)}$ and $\mathrm{d}\mathcal{B}_t^{(2)}$ are independent standard $d$-dimensional Brownian motions.[7] Using Itô's formula (see (43) for details), one then can easily derive the simplified ODEs for the system $\nu(t; \lambda_j) \stackrel{\text{def}}{=} (\rho_j^2(t), \xi_j^2(t), \chi_j(t))^\top$.

## 4   Deriving the scaling laws of (Gen-Mom-SGD) on the PLRF model

**Random matrix theory of the PLRF.**   By solving these ODEs for $\mathcal{P}$ in terms of their initial data, we can represent the curve $\mathcal{P}$ entirely as a function of the spectral data $\left(\lambda_j, \rho_j^2(0)\right)_1^d$. A mathematically convenient representation for this data is a pair of pure-point measures

$$\widetilde{\mu}_{\mathcal{F}}(\mathrm{d}x) = \sum_{i=1}^d \delta_{\lambda_j}(\mathrm{d}x)\rho_j^2(0) \quad \text{and} \quad \widetilde{\mu}_{\mathcal{K}}(\mathrm{d}x) = \sum_{i=1}^d \delta_{\lambda_j}(\mathrm{d}x)\lambda_j^2, \tag{8}$$

in terms of which $\mathcal{P}$ can be represented as a solution of an integral equation (see (36), (56), (57)).

Random matrix theory (RMT) gives a prediction for these measures $(\mu_{\mathcal{F}}, \mu_{\mathcal{K}})$, using the so-called *deterministic equivalent*; for details, see (33). Deriving such an equivalent is a textbook tool in RMT, but the proof of equivalence in the case of PLRF falls outside of textbook RMT (see for example [62]). To limit the scope of the theoretical analysis, all the scaling law statements that follow are proven for the deterministic equivalent; the numerical agreement between $\mu_{\mathcal{F}}$ and $\widetilde{\mu}_{\mathcal{F}}$ is excellent (see Fig. 3c).

**Scaling laws for SGD on the PLRF.** In [80], using the deterministic equivalent $(\mu_{\mathcal{F}}, \mu_{\mathcal{K}})$, the authors show for SGD,

(deterministic loss curve) $\mathcal{P}(t) \asymp \hat{\mathcal{F}}(\hat{\vartheta}(t)) + \gamma_2 \hat{\mathcal{K}}_{pp}(\hat{\vartheta}(t))$, where $\hat{\mathcal{F}}(t) = \hat{\mathcal{F}}_{pp}(t) + \hat{\mathcal{F}}_{ac}(t) + \mathcal{F}_0$,

where $\hat{\vartheta}(t) \stackrel{\text{def}}{=} 1 + 2\gamma_2 Bt$. Here $\hat{\mathcal{F}} \circ \hat{\vartheta}$ can be viewed as the bias term and $\hat{\mathcal{K}}_{pp} \circ \hat{\vartheta}$ as the variance due to the stochastic gradients. Each of these terms has an interpretation and an explicit scaling law:

| Component | Symbol | Scaling Law | Contributing Part of $\mu$ |
|---|---|---|---|
| **Population bias** | $\hat{\mathcal{F}}_{pp}(t)$ | $\asymp t^{-(2\alpha+2\beta-1)/(2\alpha)}$ | Spikes (pure point part) in $\mu_{\mathcal{F}}$ |
| **Model capacity** | $\mathcal{F}_0(t)$ | $\asymp d^{-(2\alpha+(1-2\beta)_+)}$ | Irreducible loss level, $\mu_{\mathcal{F}}(\{0\})$ |
| **Embedding bias** | $\hat{\mathcal{F}}_{ac}(t)$ | $\asymp d^{-1}t^{-1+1/(2\alpha)}$ | Bulk (absolutely continuous part) of $\mu_{\mathcal{F}}$ |
| **Variance** | $\hat{\mathcal{K}}_{pp}(t)$ | $\asymp d^{(1-2\alpha)_+}t^{-2+1/(2\alpha)}$ | Pure point part of $\mu_{\mathcal{K}}$ |

*4 Distinct Phases.* For different regions of $(\alpha, \beta)$, one or more of these four terms may be fully dominated by the others, yielding 4 different phases. See also Fig. 5 as an example. The different components of the loss curves do not change across the algorithms, but the scaling exponents $(\sigma, \tau)$ vary. The phase boundaries are determined by major quantitative and qualitative changes in the problem geometry.

---

[7]We have also used two Brownian motions for additional simplification of the analysis; this would correspond to using two independent stochastic gradient estimates in the two lines of (Gen-Mom-SGD).

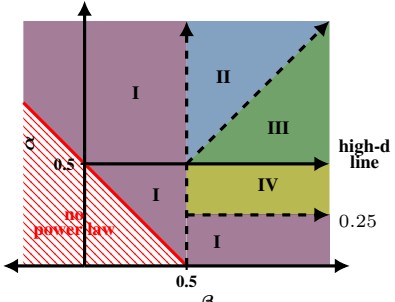

| **Dominant terms** | |
|---|---|
| **Phase I:** | $\hat{\mathcal{F}}_{pp}(t) + \mathcal{F}_0$ |
| **Phase II:** | $\hat{\mathcal{F}}_{pp}(t) + \hat{\mathcal{F}}_{ac}(t) + \mathcal{F}_0$ |
| **Phase III:** | $\hat{\mathcal{K}}_{pp}(t) + \hat{\mathcal{F}}_{ac}(t) + \mathcal{F}_0$ |
| **Phase IV:** | $\hat{\mathcal{F}}_{pp}(t) + \hat{\mathcal{K}}_{pp}(t) + \mathcal{F}_0$ |

Each phase is characterized by different dominant terms in the loss function $\mathcal{P}(t)$. The terms $\hat{\mathcal{F}}_{pp}(t)$, $\hat{\mathcal{K}}_{pp}(t)$, $\hat{\mathcal{F}}_{ac}(t)$, and $\mathcal{F}_0$ represent different components that contribute to the overall loss behavior, with their relative importance varying across the parameter space.

Figure 4: Phase diagram and corresponding dominant loss terms.

The high-dimensional line, which occurs where $2\alpha = 1$, separates gradient norms that grow with $d$ from constant ones. The line $2\beta = 1$ determines if the $\|\theta_0 - \theta^\star\|^2$ grows with $d$. The line $\alpha = \beta$ determines if the target complexity or data complexity is higher, which determines the relevance of gradient noise. Finally $\alpha = \frac{1}{4}$ determines where the gradient noise becomes high-dimensional and has aspects which are not power law.

**Scaling laws for momentum methods.** Our main result is the extension of these scaling laws to SGD-M, DANA-constant and DANA-decaying in the small batch setting.

**Theorem 4.1.** *(Summarized Version of Thm G.1 (SGD-M), Thm H.3 (DANA-constant), Thm I.2, (DANA-decaying)). Suppose that $(\alpha, \beta)$ are not on any critical line, that $2\alpha + 2\beta > 1$, that $\alpha > \frac{1}{4}$, and that $\beta \le \alpha + 1$. Suppose that batch size $B$ is fixed independent of $d$. Consider DANA-constant, DANA-decaying, SGD, and SGD-M. Define the time change*

$$\vartheta(t) \stackrel{def}{=} \begin{cases} 1 + 2(\gamma_2 + \frac{\gamma_3}{\delta})Bt, & \text{(SGD-M)}, \\ 1 + 2\gamma_2 Bt + \left(\int_0^t \sqrt{\gamma_3(s)B}\,\mathrm{d}s\right)^2, & \text{(DANA)}. \end{cases} \tag{9}$$

*Then, provided the algorithm is stable[8], which is to say that for some constant $c$ sufficiently small, independent of $d$ (and where $\mathrm{tr} = \sum_{j=1}^d j^{-2\alpha} \asymp d^{(1-2\alpha)_+}$)*

$$\begin{cases} \delta < 2, (\gamma_2 + \frac{\gamma_3}{\delta}) \le c\min\{\frac{1}{B}, \frac{1}{\mathrm{tr}}\}, & \text{(SGD-M)}, \\ c\delta > 1, \gamma_2 \le c\min\{\frac{1}{B}, \frac{1}{\mathrm{tr}}\}, \tilde{\gamma}_3 d^{-\kappa_2} \le c\gamma_2, & \text{(DANA) with } \kappa_3 \ge \frac{1}{2\alpha}, \\ c\delta > 1, \gamma_2 \le c\min\{\frac{1}{B}, \frac{1}{\mathrm{tr}}\}, \tilde{\gamma}_3 d^{-\kappa_2} \le c\gamma_2 d^{2\alpha(\kappa_3 - \frac{1}{2\alpha})}, & \text{(DANA) with } \kappa_3 < \frac{1}{2\alpha}, \end{cases} \quad \text{(stability)}$$

*one has the following scaling law where $\gamma = \gamma_2$ for DANA and $\gamma = \gamma_2 + \frac{\gamma_3}{\delta}$ for SGD-M*

$$\boxed{\mathcal{P}(t) \asymp \hat{\mathcal{F}}(\vartheta(t)) + \gamma \hat{\mathcal{K}}_{pp}(\vartheta(t)).} \tag{10}$$

*For DANA–decaying, we have the following additional requirements. We only consider $2\alpha > 1$, $\gamma_2$ bounded below, and $(1/2\alpha) < \kappa_3 < 1$.*

We remark that DANA-decaying/constant appear to be the most interesting (and optimal) DANA cases (see Fig. 6). From (10), it is simple to produce explicit scaling laws for DANA

$$\mathcal{P}(t; d) \asymp \underbrace{d^{-\tau_1} t^{-\sigma_1}}_{\text{population bias}} + \underbrace{d^{-\tau_2}}_{\text{model capacity}} + \underbrace{d^{-\tau_3} t^{-\sigma_2}}_{\text{embedding bias}} + \underbrace{d^{-\tau_4} t^{-\sigma_3}}_{\text{variance}}, \tag{11}$$

see for example Figure 5. The parameters $\sigma_i$ and $\tau_i$ are explicit, depend on the algorithm, and vary continuously in $(\alpha, \beta)$. We relegate the proofs to Sec. G, H, I and specific exponents to Table 4.

**Understanding stability.** The stability conditions listed in Theorem 4.1 are essentially sharp, in that we can show the loss curves are unbounded in $d$ if the hyperparameters have lower bounds that are on the same order as these upper bounds.

---

[8]Stability is proved in Cor. G.1 for SGD-M, Cor. H.2 for DANA-constant, Prop. I.2 for DANA-decaying.

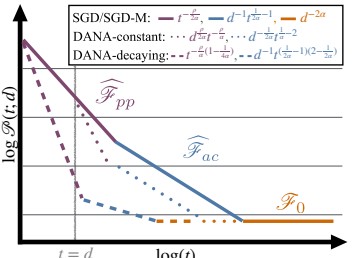

*Figure details:* Here $\rho \overset{\text{def}}{=} 2\alpha + 2\beta - 1$. Population bias, $\hat{\mathcal{F}}_{pp}(\vartheta(t))$ dominates at small times, then the loss slows down from embedding bias, $\hat{\mathcal{F}}_{ac}(\vartheta(t))$, and then finally reaches the model capacity, $\mathcal{F}_0(t)$. In this phase, the gradient noise (variance, $\hat{\mathcal{K}}_{pp} \circ \vartheta$) is always dominated by $\hat{\mathcal{F}}_{pp} \circ \vartheta$, unlike Phases III/IV. When $t < d$, DANA-constant behaves like SGD/SGD-M. For all phases, see Fig. 7, 12 for phase diagrams and Fig. 13,14,15 for loss curves.

Figure 5: **Example of loss curves (Phase II, $\alpha > \beta > 0.5$).**

In the case of SGD-M, above the high-dimensional line ($2\alpha > 1$), the effective learning rate $\gamma_2 + \frac{\gamma_3}{\delta}$ can remain constant, but below it must scale with the parameter count $d$. This transition is precisely captured by $\text{tr} \asymp \sum_{j=1}^{d} j^{-2\alpha}$.

For the DANA class, the simplest choice of $\gamma_3$ is to take it constant, (i.e. $\kappa_3 = 0$, DANA-constant). The largest constant stable choice for $\gamma_3$ is $d^{-1}$ (see also a related algorithm [94]). This has a key limitation: with such small $\gamma_3$, the momentum term needs $O(d)$ steps to reach the gradient magnitude and to have any effect on the behavior of the loss. Thus, for early iterations, $t \leq d$, DANA-constant and SGD exhibit similar scaling behavior.

One can resolve this, and produce a faster algorithm by observing that after $t$ iterations, only part of the $d$-dimensional feature space is being used, i.e. there is an *effective dimension* of $\check{K}$ at iteration $t$ [14, 61, 101, 110]. Quantitatively,

$$(\text{effective dimension at iteration } t) \asymp \max\{j : \lambda_j(\check{K}) > 1/(tB)\} \asymp (tB)^{-1/(2\alpha)}.$$

If we replace the fixed dimension parameter $d$ in DANA-constant with $(tB)^{-1/(2\alpha)}$, this yields the DANA-decaying algorithm (see also [107]) with the smallest possible $\kappa_3$. Fig. 3 shows DANA-decaying with $1/(2\alpha)$ appears to be optimal on PLRF when varying $(\kappa_2, \kappa_3)$; see also Fig. 6.

The choice of $\delta$ in $\Delta(t) = \delta(1+t)^{-1}$ must be chosen sufficiently large to guarantee acceleration and stability. We summarize the recommended DANA-constant and DANA-decaying hyperparameters for PLRF in the table below, where $\text{tr} = \sum_{j=1}^{d} j^{-2\alpha}$.

| Algorithm | | Hyperparameters $\left(\frac{\delta}{2} + \frac{\kappa_3}{4} > (2 - \kappa_3)\max\{\frac{2\alpha+2\beta-1}{\alpha}, 4 - \frac{1}{\alpha}\}\right)$ |
|---|---|---|
| **DANA-constant** | $2\alpha > 1$ | $\kappa_1, \kappa_3 = 0, \kappa_2 = 1, \gamma_2 = 1/(2\,\text{tr}), \tilde{\gamma}_3 = 1/5 \times \gamma_2$ |
| | $2\alpha < 1$ | $\kappa_1 = 1 - 2\alpha, \kappa_2 = 1 + \kappa_1, \kappa_3 = 0$ |
| **DANA-decaying** | $2\alpha > 1$ | $\kappa_1, \kappa_2 = 0, \kappa_3 = 1/(2\alpha), \gamma_2 = 1/(2\,\text{tr}), \tilde{\gamma}_3 = 1/5 \times \gamma_2$ |

**Hyperparameters beyond PLRF.** In settings where $(\alpha, \beta)$ are unknown or meaningless, we recommend setting $\gamma_2$ to a stable SGD learning rate, $\gamma_3(t) = \gamma_2 \times (1+t)^{-\kappa_3}$ and $\delta = 8$. Then, sweep over $\kappa_3$; the minimum $\kappa_3$ where DANA-decaying converges appears optimal (Fig. 2c & 20).

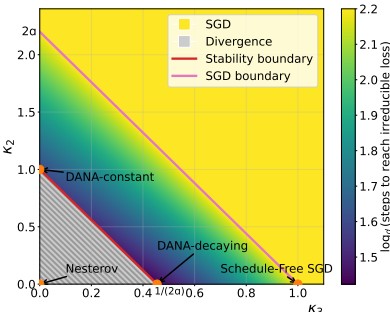

*Full DANA class.* There are other stable scaling rules in the DANA class (above red line). DANA-decaying w/ $\kappa_3 = \frac{1}{2\alpha}$ is optimal (e.g., fewest iterations to optimum, best loss exponent) amongst the whole class. **(left, above high-d line, $\kappa_1 = 0$, $B = 1$, $\alpha = 1.1$):** Plot of $\log_d$(time to reach irreducible loss). DANA stability boundary (red) at $\kappa_2 = -2\alpha\kappa_3 + 1$ with divergence below; DANA takes same number of iterations as SGD at (pink) line, $\kappa_2 \geq 2\alpha(1 - \kappa_3)$. Darker color = smaller number of steps. Iterations to reach irreducible loss, $d^{4\alpha^2/(4\alpha-1)} \leq d^{\alpha+1/2} \leq d^{2\alpha}$ (DANA-decay $\leq$ DANA-constant $\leq$ SGD). Stochastic Nesterov does not converge [38].

Figure 6: **Full DANA class, time to reach irreducible loss.**

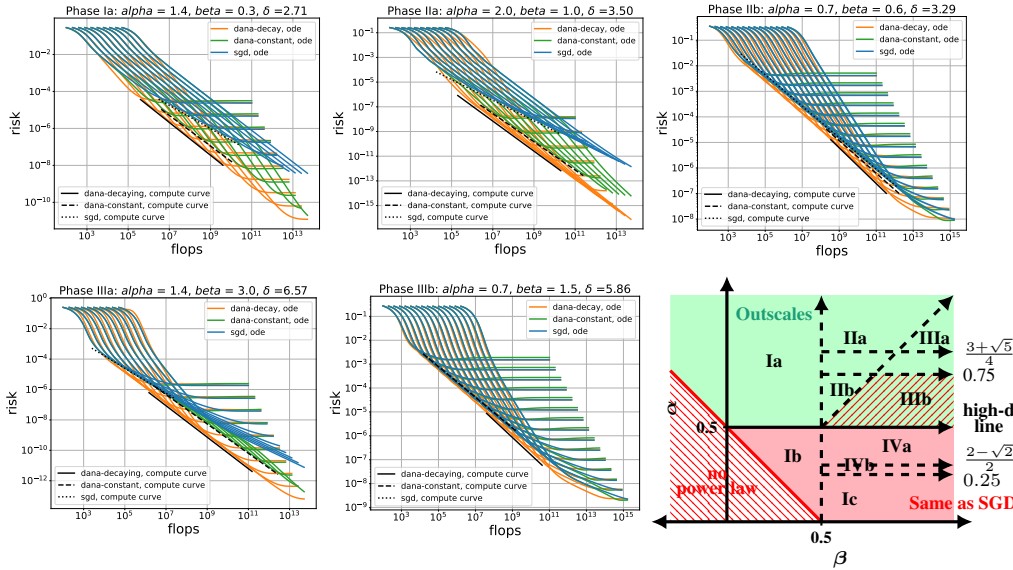

Figure 7: **Comparison of SGD, DANA-constant, and DANA-decaying with compute-optimal curve predictions.** Numerical set-up: $d = 100 \times 2^i$, $i = 1, \ldots, 15$; Simplified ODEs (43) plotted for the scaling-law-equivalent model. DANA-decaying outscales in all phases (Phase Ia, II, III) where $2\alpha > 1$. DANA-constant outscales SGD in all phases $2\alpha > 1$ except Phase IIIb. Predictions for compute-optimality loss exponents match empirical results. (see Sec. M for details).

## 5   Using Momentum to outscale SGD

**SGD-M fails to outscale.** The typical approach to momentum is to use a constant momentum $\Delta(t) \equiv \delta$, in practice usually set to $1 - \delta = 0.9$. Regardless of the choice of fixed $\Delta$, SGD-M produces the same scaling laws as SGD. In particular, most 'speed up' of SGD-M can be attributed to a larger effective learning rate for SGD, resulting in improved scaling law constants, but not a change in the loss exponent. There is an equivalence in risk dynamics when $\gamma_2^{\text{SGD}} = \gamma_2^{\text{SGD-M}} + \gamma_3^{\text{SGD-M}}/\delta$; see Rem. G.2 for details and Fig. 16 and 24 for empirical equivalence on PLRF and LSTMs respectively.

**DANA-constant outscales SGD for most** $(\alpha, \beta)$ **with** $2\alpha > 1$**.** The limitation of DANA-constant's small $\gamma_3$ learning rate is that the momentum term requires at least $d$ iterations of the algorithm to become noticable in the loss dynamics. Thus, for $t \leq d$, DANA-constant and SGD have the same scaling law (see Fig. 1a where gray line indicates $t \asymp d$). If one is able to reach $\mathcal{F}_0$ before $d$ (which occurs for $2\alpha > 1$), DANA-constant will outscale SGD in training regime $t = d^\ell$ with $1 < \ell < 2\alpha$. We note that once $\ell = 2\alpha$, no algorithm outscales SGD, as the irreducible loss level is reached.

**DANA-decaying outscales DANA-constant for all** $(\alpha, \beta)$ **with** $2\alpha > 1$**.** For any regime $t = d^\ell$ with $0 < \ell < 2\alpha$, DANA-decaying will outscale both SGD and DANA-constant, provided the former is not already at the irreducible loss level $\mathcal{F}_0$. Hence DANA-decaying will always be both more *sample efficient* and *compute efficient* than SGD and DANA-constant. One may look further at the full DANA class of algorithms (Fig. 6), which may potentially contain other interesting scaling rules.

Moreover, many stochastic algorithms fall into Gen-Mom-SGD – Schedule-Free SGD [33], Nesterov [75], AcSGD [94], Accelerated SGD [52, 61] (see Tab. 3 for param. comparison). For these, deterministic ODEs (22) exactly describe the risk evolution. We give heuristics for the scaling laws of these algorithms (Sec. B.5). See Tab. 2 for sample complexity comparison and Fig. 10 for scaling behavior, in which DANA-decaying outperforms all (including Adam with cosine decay, see Fig. 1c).

In principle, one could also look at *unstable* scaling rules, which are stable in one training regime but eventually diverge. For example, one could sweep over the $\gamma_3$ in DANA-constant to find the $\gamma_3$ that minimizes $\mathcal{P}(t; d)$; this produces a schedule $\gamma_3(t; d)$, which on the PLRF is exactly DANA-decaying with $\kappa_3 = 1/(2\alpha)$ (see Fig. 3b & Section J.3). See also Fig. 3a on PLRF and Fig. 2a on LSTM for a related sweep over $\kappa_3$.

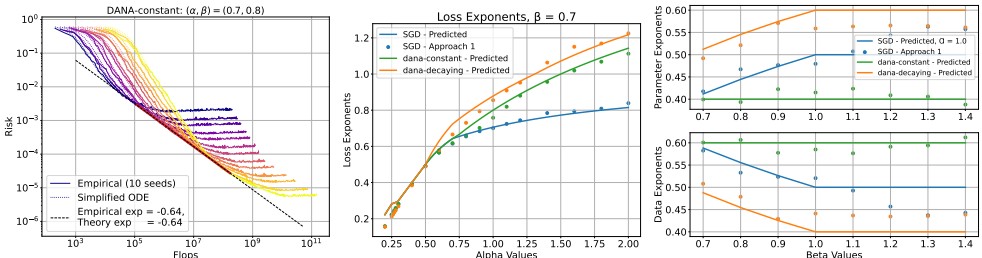

Figure 8: **Loss curves match almost exactly between empirical PLRF experiments and ODE solutions. Compute-optimal loss exponents match approximately.** *(left)* Loss curves for simplified ODEs (7) closely match empirical PLRF across 13 $d$-values up to $d = 12,800$. Empirical loss exponent fit by Chinchilla Approach 1 [50]. *(middle)* Loss exponents $\eta$ in $\mathscr{P}(\mathfrak{f}/d^\star; d^\star) = \mathfrak{f}^\eta$ for fixed $\beta = 0.7$ across $\alpha$'s. Theoretical predictions (solid lines, see Tab. 5,6,7) match empirical (dots) within 0.09. *(right)* Compute-optimal parameter exponents $\xi$ in $d^\star(\mathfrak{f}) = \mathfrak{f}^\xi$ match within 0.13 (top) and data exponents, $\zeta$ (bottom), for fixed $\alpha = 1$ across $\beta$ values. See extensive experiments Sec. K across 84 values of $(\alpha, \beta)$ + discussion of error sources. Fig. details in Sec. M.

**Compute-optimal regime.** One natural training regime for $d$ is the *compute-optimal* training regime [16, 50, 63], where for a given compute budget $\mathfrak{f}$ measured in flops, one chooses $d^\star(\mathfrak{f})$ to minimize the loss. We measure $\mathfrak{f}$ via:

$$\text{Compute (flops } \mathfrak{f}) = (\text{iterations of alg. } (t) \ \times \text{ batch size } (B)) \ \times \text{ parameters } (d). \quad (12)$$

We plot the loss curve $\mathscr{P}(\theta_t; d) = \mathscr{P}(t; d) = \mathscr{P}(\frac{\mathfrak{f}}{d}; d)$ as a function of flops, and then we solve for the compute-optimal parameter size $d^\star(\mathfrak{f}) = \arg\min_d \mathscr{P}(\frac{\mathfrak{f}}{d}; d) = \mathfrak{f}^\xi$. With access to the explicit functional form of the loss curve (11), it is straightforward to find $d^\star$. We denote $\xi$ the *parameter exponent* and $\eta$ the *loss exponent*. Here $\mathscr{P}(\frac{\mathfrak{f}}{d^\star}; d^\star) = \mathfrak{f}^{-\eta}$ is known as the *compute-optimal curve*. The *data exponent* is $1 - \xi$ since iterations times batch equals amount of data used.

Given the form of the loss curves (11), compute-optimality must occur at an intersection point of $\hat{\mathcal{F}}_{pp}, \hat{\mathcal{F}}_{ac}, \mathcal{F}_0$, and $\hat{\mathcal{K}}_{pp}$. Thus, the phases are further broken down depending on the tradeoff location of compute-optimality. We derive compute-optimal frontiers and present the exponents in Tab. 5,6,7, Sec. E for proof details. See also Fig. 12 (phase diagrams) and Fig. 13,14,15 cartoon plots of loss curves and compute-optimality. Fig. 9 gives a detailed example of the shape of the risk curves. Predictions of exponents $(\zeta, \eta, \xi)$ match theory - see Fig. 8, Fig. 26-33. For specific details about the different phases and algorithms, see Sec. E.5.

**Compute-optimality main takeaways.** The empirical Chinchilla law [50] showed the optimal parameter count scales like $d^\star(\mathfrak{f}) \asymp \mathfrak{f}^{1/2}$ on large language models while [80] observed theoretically that SGD on PLRF reproduces this behavior in Phase III. However, for $2\alpha > 1$, DANA-decaying is *never* compute-optimal with $d^\star(\mathfrak{f}) \asymp \mathfrak{f}^{1/2}$: depending on the phase, the compute-optimal training regime of DANA-decaying requires undertraining or overtraining relative to the Chinchilla law.

Furthermore, the compute-optimal tradeoff point can change based on the choice of algorithm. For example, in Phase IIa/IIIa, for SGD it is compute-optimal to stop training early, when the optimization enters the $\hat{\mathcal{F}}_{ac}$ regime (which is slower to optimize). However, for DANA-decaying, it is compute-optimal to continue training, and in fact, to continue training to the irreducible loss level $\mathcal{F}_0$ because DANA-decaying is substantially more sample/compute-efficient than SGD in this regime. Moreover, even when the tradeoff point is the same as SGD, such as in Phase Ia, DANA-decaying attains better loss than SGD for the same compute or sample budget.

**LSTM results.** We train 2-layer LSTMs (Fig. 2) on the C4 language dataset [84] and co-scale the embedding and hidden dimensions to sweep model sizes. This in-

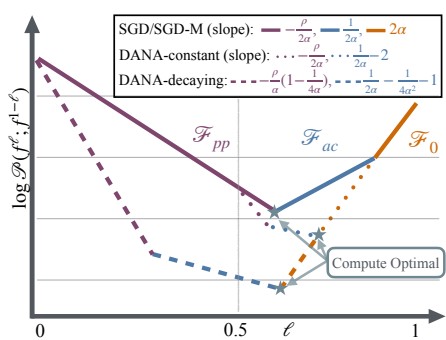

Figure 9: **Compute-optimal scaling laws (Phase IIa,** $(\alpha > (3 + \sqrt{5})/4, \alpha > \beta > 0.5)$**).** We reparameterize Fig. 5 by fixing the compute budget $\mathfrak{f}$ and parameterizing the x-axis with $\ell \in [0, 1]$, so that $d = \mathfrak{f}^{1-\ell}$ and $t = \mathfrak{f}^\ell$. This shows the relationship between compute budget $\mathfrak{f}$ and optimal parameter count.

duces a power-law regime for the compute-optimal frontier on intermediate model sizes with all $R^2 \geq 0.984$, similar to Fig 7 in [54]. For DANA-decaying, the loss exponents (Fig. 22) vary smoothly with $\kappa_3$ and closely match PLRF behavior (Fig. 6): diverging for $\kappa_3 < 0.6$, outscaling SGD for $0.6 \leq \kappa_3 \leq 0.9$, and matching SGD when $\kappa_3 \geq 1.0$ where $\kappa_3 = 1.0$ is Schedule-Free SGD. We show equivalence for SGD-M and SGD in Fig. 24. For details see Sec. L.

**Conclusion, limitations and future work.** We have shown that DANA outscales SGD on PLRF in the $2\alpha > 1$ regime by properly scaling Nesterov-type momentum in a data- and dimension-dependent way. We validated Theorem 4.1 with extensive PLRF experiments; moreover, PLRF acts as a useful proxy for LSTMs on text data where theoretical predictions for loss exponent improvements hold. A full convergence argument beyond quadratics for DANA-decaying would be desirable, e.g. theory for cross-entropy might suggest different momentum scaling strategies. It is an open question if outscaling is possible for either the $2\alpha < 1$ case or for $d$-dependent batch sizes. Additional limitations of our analysis include the use of fixed features which does not incorporate any kind of feature learning, as well as deterministic learning dynamics based on an ODE description of the loss evolution.

Additionally, while we compare DANA against other non-adaptive momentum methods, most real-world problems benefit from adaptive methods such as Adam. Hence it would be interesting to have an analysis of DANA combined with Adam or other preconditioned methods. Finally, in the LSTM setting, the exact meaning of $\alpha$ is not clear, although empirically $\alpha = 0.71$ (corresponding to $\kappa_3 = 0.7$) appears near optimal. Defining and measuring this $\alpha$ on real-world problems, particularly determining whether $2\alpha > 1$, is an important direction for future work.

# Acknowledgments and Disclosure of Funding

D. Ferbach is supported by Fonds de Recherche du Québec (FRQ) (DOI assigned: https://doi.org/10.69777/363626). G. Gidel is a CIFAR AI Chair, he is supported by a Discovery Grant from the Natural Science and Engineering Research Council (NSERC) of Canada and a Google x Mila research grant. C. Paquette is a Canadian Institute for Advanced Research (CIFAR) AI chair, Quebec AI Institute (Mila) and a Sloan Research Fellow in Computer Science (2024). C. Paquette was supported by a Discovery Grant from the Natural Science and Engineering Research Council (NSERC) of Canada, NSERC CREATE grant Interdisciplinary Math and Artificial Intelligence Program (INTER-MATH-AI), Google x Mila research grant, Fonds de recherche du Quebec Nature et technologies (FRQNT) NOVA Grant, and CIFAR AI Catalyst Grant. Research by E. Paquette was supported by a Discovery Grant from the Natural Science and Engineering Research Council (NSERC). This research was enabled in part by compute resources provided by Mila (mila.quebec). Additional revenues related to this work: C. Paquette has 20% part-time employment at Google DeepMind.

The authors would like to thank Jeffrey Pennington, Lechao Xiao, and Atish Agarwala for their careful reading and helpful feedback that improved the paper.

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

# Supplementary material

**Broader Impact Statement.** The work presented in this paper is foundational research and it is not tied to any particular application. The primary set-up is on a simple well-studied random features model with synthetic data and solved using commonly deployed algorithms – variants of stochastic gradient descent with momentum. We present (theoretical) compute-optimal curves for this model. While a common positive foreseeable impact of algorithms with better scaling properties is that it would allow more efficient model training and hence lower energy consumption of AI, it is important to acknowledge that, many times across history, cost-lowering technological improvements have led to increase in consumption due to Jevons paradox. Regarding our experiments, we include results on language models trained on a public language dataset that show the theoretical behavior holds on more practical settings. We do not release any pretrained models but because the dataset we used is based on common crawl which has potential issues of fairness, harmfulness, accountability, and transparency we may be contributing to reinforce these issues by supporting the making of the current C4 dataset as a standard in NLP, obfuscating these concerns. Finally, since our main results are theorems regarding scaling laws, we anticipate that one potential negative impact of such a line of research is that fundamental advance in scaling laws will advantage large entities that have more computational resource, thus enhancing centralization of technological power.

**Outline of the paper.** The remainder of the article is structured as follows:

1. In Section A we discuss related works and compare previous convergence rates to ours.

2. In Section B we provide more details on the algorithms under study, i.e., SGD, SGD-M, DANA-constant, DANA-decay. We additionally show that multiple existing algorithms can be re-framed within the class (Gen-Mom-SGD). We then use our results to precisely conjecture their scaling laws.

3. In Section C we derive the exact ODEs for the loss along with the exact Volterra equation. We discuss the simplification of this ODE and the resulting Volterra equation. We finally give some background on Volterra equations and how to reduce their complexity.

4. In Section D we derive the random matrix analysis of the deterministic equivalent measures $\mu_{\mathcal{F}}$, $\mu_{\mathcal{K}}$. Previous results from [80] estimated the deterministic equivalent on a contour enclosing the spectrum of the data covariance matrix. We reformulate and strengthen these on the real line and provide general results for the integration of well-behaved real valued functions against $\mu_{\mathcal{F}}$, $\mu_{\mathcal{K}}$.

5. In Section E we derive the compute-optimal scaling laws for SGD-M, DANA-constant and DANA-decaying.

6. In Section F we analyse SGD with a general learning rate schedule.

7. In Section G we derive the scaling laws for SGD-M, and prove that they are identical to SGD with a precise correspondence between the learning rates of these two algorithms.

8. In Section H we analyse the DANA-constant algorithm. We use Frobenius method to obtain asymptotic estimates for the solutions of the simplified ODE system (43) in Sections H.2 to H.4. We use these estimates to derive stability conditions on the hyperparameters in sections H.8 and H.9 and compute the forcing and kernel functions.

9. In Section I we analyse DANA-decaying above the high-dimensional line. We give estimates on the solutions of (43) in Section I.1. We use these to derive the forcing and kernel functions, and the scaling laws. Finally, we discuss in Section I.5 a heuristic generalization to the algorithm class (DANA) with stability conditions and associated scaling laws.

10. In Section J we study whether non-stable learning rates can accelerate the dynamics early in training. Especially we show that sweeping over the $\kappa_1$ hyperparameter in DANA-constant recovers the DANA-decaying schedule.

11. In Section K we discuss experimental details on the PLRF model and show numerical agreement between empirical and theoretical PLRF behavior.

12. In Section L we discuss experiments on LSTMs on a language dataset.

13. In Section M we report further experimental details for figures.

**Notation.** We use $\mathscr{P}(\theta_t) = \mathscr{P}(t)$ when we want to emphasize the iteration counter $t$. We say $\mathscr{A}(t, v, d) \sim \mathcal{A}(t, v, d)$ for functions $\mathscr{A}(t, v, d), \mathcal{A}(t, v, d) > 0$ if for every $\varepsilon > 0$ and for all admissible $v$ and $d$, there exists $t_0, d_0$ such that for all $d > d_0$ and $t \geq t_0$

$$(1 - \varepsilon)\mathcal{A}(t, v, d) \leq \mathscr{A}(t, v, d) \leq (1 + \varepsilon)\mathcal{A}(t, v, d).$$

For some $v \in \mathbb{N}^*$, we denote the diagonal matrix $D = \mathrm{Diag}(j^{-2\alpha} : 1 \leq j \leq v)$.

For $A, B$ two matrices, comparisons of the form $A \leq B$ are understood coordinate-wise. Additionally, $\|A\|$ denotes the matrix of the absolute values of coordinates of $A$. For $x \in \mathbb{R}_+$, $\sqrt{x} \geq 0$ denote the positive square root of $x$ and $\sqrt{-x} \stackrel{\text{def}}{=} i\sqrt{x}$. $\mathbb{H} \stackrel{\text{def}}{=} \{z \in \mathbb{C}, \, \mathrm{Im}(z) > 0\}$ denotes the complex upper-half plane. For $\eta > 0$ and $u \in \mathbb{R}$ we denote the Poisson kernel $\mathrm{Pois}_\eta(u) \stackrel{\text{def}}{=} \frac{1}{\pi}\frac{\eta}{u^2+\eta^2}$. For $x \in \mathbb{R}_+$, we denote $\overline{x} = \max\{1, x\}$. For $x \in \mathbb{R}$, we use $x_+ \stackrel{\text{def}}{=} \max\{x, 0\}$.

**Detailed proof sketch.** We finally provide below a detailed proof sketch of our main result Theorem 4.1

1. The first step is to derive an evolution equation for the quadratic risk using the raw algorithm updates Equation (Gen-Mom-SGD).

   - We adopt in Section C.1 a continuized analysis in Equation (Gen-Mom-SGD) by replacing the update times by the jumps of a Poisson process of rate 1 (Equation (18)). This allows us to reduce the analysis to ODE systems without sending the learning rates to 0.
   - In Section C.1.1 we derive ODEs for the projected risk on each eigenvalue direction of the covariance matrix for two algorithms: the original one in Section C.1.1 and the coin-flip algorithm in Section C.2.2. The coin-flip algorithm uses two independent Poisson processes for the momentum and parameter updates in Equation (Gen-Mom-SGD) which gives rise to much simpler ODEs.
   - The ODEs obtained are theoretically solvable but a lot of small high-order terms make the analysis difficult. Hence we drop high-order terms in $\lambda$ in the exact and coin-flip ODEs to obtain the 'simplified ODEs' (Section C.2.2). We prove in Section C.2.1 that the 'simplified ODE' exactly represents the evolution of the SDEs presented in Equation (7).
   - Finally, using these ODEs, we can derive a Volterra equation for the full quadratic risk $\mathcal{P}(t) = \mathcal{F}(t) + (\mathcal{K} * \mathcal{P})(t)$ where $\mathcal{P}(t)$ is the risk, $\mathcal{F}(t)$ is the forcing function, $\mathcal{K}(t, s)$ is the kernel function. This is done in Sections C.1.2 and C.2.3.
     The rest of the proof focuses on computing asymptotics for $\mathcal{F}$ and $\mathcal{K}$ (Equation (55)) by quantifying the empirical distribution of the covariance matrix spectrum and solving the ODEs.

2. Quantifying the spectrum of the data covariance matrix $\hat{K}$.

   - We first rewrite $\mu_F, \mu_K$ in Equation (55) as integrals against two measures $\mu_{\mathcal{F}}, \mu_{\mathcal{K}}$ which are defined through the deterministic equivalent in Equation (63).
   - Using Random Matrix Theory and previous results from [80], we provide upper and lower bounds on the deterministic equivalent measures $\mu_F, \mu_K$ in Sections D.1 and D.2.
   - In Sections D.3 and D.4 we use these bounds to define diverse components of the forcing function and kernel function (Equations (66) and (68)), formalized in Propositions D.13 and D.14.

3. We then solve the 'simplified ODE' Equation (54) for each class of algorithm in Equation (Gen-Mom-SGD) separately.

   - For SGD-M we solve the ODE in Section G.1. These are the simplest as the ODE has constant coefficients, and the solutions are exponentials.
   - For DANA-constant, the ODE has time-varying coefficients. We employ Frobenius theory to obtain asymptotic solutions of the ODE in Section H.2 to Section H.6.
   - For DANA-decaying, we employ orthogonal polynomial theory to solve the ODEs in Section I.1.

4. Finally, we need to solve the Volterra equation for the risk.

- For each algorithm to remain stable, the kernel norm has to be smaller than 1. This leads to stability conditions on the learning rates for SGD-M (Corollary G.1), DANA-constant (Corollary H.2) and DANA-decaying (Proposition I.5).
- Using this stability condition we can simplify the Volterra equation as $\mathcal{P} \asymp \mathcal{F} + \mathcal{K}$ using Kesten's lemma (Propositions G.8, H.20 and I.5).
- We then compute $\mathcal{F}, \mathcal{K}$ for each algorithm by integrating the previous solutions against $\mu_F, \mu_K$ and obtain explicit scaling laws for the loss $\mathcal{P}$. This is done in Proposition G.4 for SGD-M, Propositions H.13, H.14 and H.19 for DANA-constant and Propositions I.1, I.3 and I.4 for DANA-decaying.
- Finally, we determine the compute-optimal regime in Section E by a comparison of the different terms in the loss.

## Contents

## A Related Work

**PLRF & Scaling laws on the PLRF.** Our work builds on the study of compute optimality in large language models, introduced by [50] and [54], who performed empirical investigations of this phenomenon. The problem formulation follows [69], which examined data-limited scaling scenarios but did not address compute optimality or algorithmic considerations. Related dynamical analyses

| | This work[1] | [DLM24][2] | [Bahri+21][3] | [MRS22][4] | [BAP24][5] | [Lin+24][6] |
|---|---|---|---|---|---|---|
| **Input dimension** | $d$ | $d$ | $d$ | $M$ | $M$ | $M$ |
| **# of features** | $v$ | $p$ | $P$ | $N$ | $N$ | - |
| **Iterations/samples** | $t$ | $n$ | $D$ | $T$ | $P$ | $N$ |
| **Capacity** | $2\alpha$ | $\alpha$ | $1+\alpha$ | $1+\alpha$ | $b$ | $a$ |
| **Source** | $\frac{2\alpha+2\beta-1}{4\alpha}$ | $r$ | $\frac{1}{2}(1-\frac{1}{\alpha})$ | $\frac{1}{2}(1-\frac{1}{\alpha})$ | $\frac{a-1}{2b}$ | $\frac{b-1}{2a}, b>1$ |
| **Target decay (in $L_2$)** | $\alpha+\beta$ | $\alpha r+\frac{1}{2}$ | $0$ | $0$ | $\frac{a}{2}$ | $\frac{b}{2}$ |

[1] [Paq+24] E. Paquette, C. Paquette, L. Xiao, J. Pennington. *4+3 Phases of Compute-optimal Neural Scaling Laws.* 2024

[2] [DLM24] L. Defilippis, B. Loureiro, T. Misiakiewicz. *Dimension-free deterministic equivalents for random feature regression.* 2024

[3] [Bahri21] Y. Bahri, D. Dyer, J. Kaplan, J. Lee, and U. Sharma. *Explaining neural scaling laws.* 2024.

[4] [MRS22] A. Maloney, D.A. Roberts, J. Sully. *A solvable model of neural scaling laws*

[5] [BAP24] B. Bordelon, A. Atanasov, C. Pehlevan. *A dynamical model of neural scaling laws.* 2024.

[6] [Lin24] L. Lin, J. Wu, S. Kakade, P. Barlett, J.D. Lee. *Scaling Laws in Linear Regression: Compute, Parameters, and Data.* 2024

Table 1: **Comparison of the source/capacity parameters across various related work.** We note this table is taken from Table 1 in [DLM24][2] and Table 4 in [Paq+24][1]. [Paq+24] also uses the same notation as this paper except samples are $r$ instead of $t$.

through gradient flow in similar settings can be found in [16]. A significant body of research has investigated scaling laws relating loss minimization to dataset size and parameter count across various settings (linear models, random features, deep networks). Notable contributions include [11], [89], and [91], which explore the "hidden-manifold" data model for *one-pass SGD*.

Scaling laws and compute-optimality for the PLRF model under one-pass SGD have been analyzed by [16, 63, 80] with an extension to feature learning in [17]. This work notably follows and extends the ideas of [80] to stochastic momentum algorithms. Particularly, [80] developed scaling laws for SGD under a deterministic equivalent and showed that there exists 4 distinct phases. They used their scaling law to find compute-optimality. Related work on ridge-regression gradient descent includes [29] and [34]. Other theoretical guarantees for scaling laws beyond PLRF, typically for gradient flow, have been established in works such as [74].

**Scaling hyperparameters in large models.** There is extensive literature studying the optimal scaling of optimizer hyperparameters in large neural networks, particularly the initialization and learning rate [39, 100, 103, 104], the batch size [70, 87, 109] and the epsilon hyperparameter in adaptive optimizers [39, 100, 105]. While the momentum hyperparameters for large models are often treated as fixed constants rather than scaled quantities, the GPT-3 model [20] set $\beta_2 = 0.95$ rather than the typical $\beta_2 = 0.99$ or $0.999$, which may improve stability with large batch sizes.

**Random features and random matrices.** Our work employs random matrix theory to examine a random features problem that, from a statistical perspective, represents the generalization error of one-pass stochastic momentum algorithms. Random matrix theory has played an increasingly large role in machine learning (see for [28] for a modern introduction).

For our random matrix analysis, we require sample covariance matrices with power law population covariance (i.e. linear random features). The analysis of sample covariance matrices precedes their usage in machine learning (see e.g. [12]). A detailed study of all parts of the spectrum of sample covariance matrices with power law population covariances appeared in [80] and has been subsequently used in [17] to study one-pass SGD and multi-pass SGD [7]. The study of ridge regression has been extensively investigated (see for e.g. [10, 24]), and the work [34] provides a complete analysis of the ridge regression problem under power law random features when $2\alpha > 1$.

There is a larger theory of nonlinear random features regression, mostly in the case of isotropic random features. For isotropic random features with *proportional dimension asymptotics*, this has been explored in works such as [71] and for some classes of anisotropic random features in [31, 68, 72]. We note that lots of the complexity of the analysis of power law random features arises from the analysis of self-consistent equations though the proof of these self-consistent equations falls outside the typical random matrix theory setting from textbooks. The use of self-consistent equations to study random matrix theory dates to [90], but the analysis of these equations (i.e, for $\check{K}$), as far as we can tell, dates to [80]). This strongly motivates non-proportional scalings (which would be inevitable in power law random features with nonlinearities); in the isotropic case, the state of the art is [51].

**Random features regression, 'source/capacity' conditions, and SGD.** A large body of kernel regression and random features literature is formulated for "source/capacity" conditions, which are power law type assumptions that contain the problem setup here, when $2\alpha > 1$ (the low-dimensional regime). For convenience, we record the parameters

$$\alpha_{\text{source}} = 2\alpha \quad \text{and} \quad r_{\text{capacity}} = \frac{2\alpha + 2\beta - 1}{4\alpha}.$$

Here we have taken $r_{\text{capacity}}$ as the limit of those $r$'s for which the source/capacity conditions hold (see Table 1). We note that in this language $r_{\text{capacity}}$ is often interpreted as 'hardness' (lower is harder), and that $r \in (0, 0.5)$, $r \in (0.5, 1.0)$ and $r \in (1.0, \infty)$ correspond to 3 regimes of difficulty which have appeared previously (see the citations below); they are also precisely the 3 Phases Ia, II, and III.

Under source/capacity conditions, the authors of [85] establish generalization bounds for random feature regression with power law structures in $2\alpha > 1$ case for one-pass SGD. These bounds were sharpened in [29] and extended in [34] (see also [98]). An earlier work [22] shows kernel ridge regression is 'minimax optimal' under various 'source-capacity conditions'. We give a comparison to these bounds in Table 2, but we note that the problem setup we have is not captured by 'minimax optimality' (in particular minimax optimality is worst-case behavior over a problem class, and our problem setup is not worst-case for the traditional source/capacity conditions). Moreover, the authors [52, 94, 107] study stochastic momentum algorithms under source/capacity conditions (see below for details).

We note that this paper is fundamentally about scaling laws, but the novel mathematical contributions could also be recast in terms of generalization bounds of one-pass stochastic momentum algorithms. For SGD, the work of [23] compares SGD to kernel ridge regression, showing that one-pass SGD can attain the same bounds as kernel ridge regression and hence is another minimax optimal method (again under 'source-capacity' conditions). See also [36] which considers similar statements for SGD with iterate averaging and [81] for similar statements for multipass SGD; see also [35, 88] which also prove the single-batch versions of these. These bounds attain the minimax-optimal rate, which are worse than the rates attained in this paper (see Table 2 for a comparison).

**Stochastic momentum algorithms.** Recent research has established convergence guarantees for stochastic classical momentum (SGD-M) in both strongly convex and non-strongly convex settings [40, 41, 76, 86, 102][9]. The latter references have established almost sure convergence results. For quadratic minimization specifically, SGD-M iterates converge linearly (though not in $L^2$) under exactness assumptions [67], while under additional constraints on stochastic gradient noise, [55] and [21] demonstrate linear convergence to a neighborhood of the solution. Batch size determination significantly impacts the convergence rate of both SGD and SGD+M. For small batch sizes, SGD+M does not necessarily outperform SGD [55, 78, 108], while some acceleration has been demonstrated for large batch sizes [15, 30, 60].

Convergence results for stochastic Nesterov's accelerated method [75] (SNAG) have been established for both strongly convex and non-strongly convex settings [6, 9, 58]. Under stronger assumptions—such as the strong growth condition [96] or additive noise on stochastic gradients [59]—convergence to the optimum at an accelerated rate can be guaranteed. As noted in [38, Thm. 7] (see also references therein), a naive implementation of stochastic Nesterov acceleration fails to converge in the non-strongly convex setting.

---

[9]This is a non-exhaustive list of work on stochastic momentum algorithms.

The absence of general convergence guarantees showing acceleration for existing momentum schemes in stochastic settings has prompted the development of alternative acceleration techniques, known as "accelerated SGD" [2, 33, 43, 44, 46, 52, 55, 58, 61, 64, 73, 94, 107], see (Gen-Mom-SGD). Recent empirical success in large transformer models indicates benefits of accelerated SGD methods [83, 109], though theoretical understanding of hyperparameter scaling with model size [78, 94] and rigorous proofs of improved compute-optimal scaling exponents remain open questions.

The idea of accelerated SGD emerged from [3, 52, 61, 75, 94]. They observed acceleration can be obtained by coupling stochastic gradient descent and another update with aggressive stepsize. Consequently one simply has to scale down the stepsize in the aggressive update to make it robust to the gradient noise. In [61] (see also [66]), the authors produce an instance dependent risk bound for an accelerated SGD algorithm using constant (dimension independent) hyperparameters (at least above the high-dimensional line). We match the hyper-paramters of Accelerated SGD [61] to our set-up in Table 3. Another popular algorithm, Schedule-free SGD [33], was proved to achieve the worst-case rate (in the sense of classical analysis of algorithms) of SGD and can also be incorporated into various other algorithms (e.g., Adam[56]). On benchmarks, it performs remarkably well. After rewriting, we can express Schedule-free SGD in the form of (Gen-Mom-SGD); see Table 3.

The algorithms AcSGD [94] and DANA in [78], are shown to exhibit acceleration for non-strongly convex quadratics by using momentum parameter $(1 + t)^{-1}$ and dimension-dependent learning rates on the order of $1/d$. Specifically, DANA [78] showed acceleration in the high-dimensional setting (samples and parameters are proportional) whereas AcSGD provides general bounds on quadratics in the setting that $2\alpha > 1$ (above the high-dimensional line) under source/capacity conditions. AcSGD achieves the rate $\mathbb{E}\left[\mathscr{P}(\theta_t)\right] \lesssim \min\{\frac{1}{t}, \frac{1}{dt^2}\}\|\theta_0 - \theta^\star\|^2 + d^{-2\alpha+(1-2\beta)_+}$. Since $\|\theta_0 - \theta^\star\|^2$ carries a $d$-dependence, it is difficult to obtain a scaling law from this. If $\|\theta_0 - \theta^\star\|^2 \asymp 1$ (as anticipated in Phase II and III), the derived rate agrees with DANA-constant (with $\kappa_2 = 1/d$) at the time $t \asymp d^{\alpha+1/2}$ that DANA-constant hits the irreducible loss level. Otherwise the derived rate is worse than the rate at any other time given for DANA-constant with $\kappa_2 = 1$; this makes sense as the AcSGD rate holds for more general data covariances and it was derived without taking into account scaling. Moreover we see that when $t \asymp d$, AcSGD behaves as $1/t$, or SGD – similar to DANA-constant $(\kappa_2 = 1/d)$. We match the hyperparameters in AcSGD with our set-up (see Table 3). Additionally, using our Theorem 4.1, We derive heuristically the scaling law for AcSGD (see Sec. B.5) which asymptotically matches DANA-constant with $\kappa_2 = 1/d$ and we perform experiments showing the close relationship between DANA-constant and AcSGD, Figure 10.

In a recent work, the authors [107] introduced *SGD with 1-memory*, a one-pass algorithm, analyzed on the (infinite) quadratic under the source/capacity constraints $(2\alpha > 1)$. In particular, the set-up is infinite dimensional $(d \to \infty)$ and does not contain the embedding matrix $W$ in PLRF. This algorithm has the same form as (Gen-Mom-SGD). The authors propose a variant of DANA-decaying (with $\kappa_3 = 1/(2\alpha)$) (see Table 3). While not proven, they heuristically expect in Phase Ia/II (signal-dominated regime) that the rate matches our $\mathcal{F}_{pp}$ rate. We prove this rate, as well as the rate for the noise-dominated regime (Phase III) (Theorem I.2). The work of [107] is based off the deterministic result which looked at power law eigenvalue distributions on the conjugate gradient algorithm, see [19]. In a concurrent work [106], the authors used constant momentum and learning rates in (Gen-Mom-SGD) to show acceleration in the infinite dimensional $(d \to \infty)$ set-up but using $\infty$-memory SGD. This would require storing multiple (in fact, infinite) vectors, each on the size of the parameter space. They propose a finite memory version that approximates the infinite version. Such a result is an interesting future direction to derive scaling laws.

In [40, 73], they showed many known algorithms could be rewritten in the form of (Gen-Mom-SGD). We include Table 3 to provide an equivalence of hyperparameters of related work with (Gen-Mom-SGD). These include stochastic Nesterov, AcSGD [94], Schedule-free SGD [33], accelerated SGD from [61], and SGD with 1-memory [107]. We perform scaling experiments with the PLRF model on these algorithms in Fig. 10 and Fig. 1c.

Lastly, the notion of *effective dimension* has been used by [14, 61, 101, 110], but never quantified directly as a time-dependent learning rate.

**Dynamical deterministic equivalents, Volterra equations and ODEs.** Using the deterministic equivalents for random matrix resolvents [47], we in turn derive deterministic equivalents for the risk curves of stochastic momentum algorithms.

| | | This work | This work |
|---|---|---|---|
| **Algorithm** | | one-pass DANA-decaying ($\kappa_3 = \frac{1}{2\alpha}$) | one-pass DANA-constant ($\kappa_2 = 1$), $n \geq d$ |
| **Risk** $\mathscr{P}(n)$ | Phase Ia | $\Theta(n^{-(\rho/(2\alpha))\cdot(2-\kappa_3)} \vee d^{-\rho})$ | $\Theta(d^{\rho/(2\alpha)}n^{-\rho/\alpha} \vee d^{-\rho})$ |
| | Phase II | $\Theta(n^{-(\rho/(2\alpha))\cdot(2-\kappa_3)} \vee d^{-2\alpha}$ $\vee d^{-1}n^{-(2-\kappa_3)(1-1/(2\alpha))})$ | $\Theta(d^{\rho/(2\alpha)}n^{-\rho/\alpha} \vee d^{-2\alpha}$ $\vee d^{-1/(2\alpha)}n^{-2+1/\alpha})$ |
| | Phase III | $\Theta(n^{-(2-1/(2\alpha))\cdot(2-\kappa_3)} \vee d^{-2\alpha}$ $\vee d^{-1}n^{-(2-\kappa_3)(1-1/(2\alpha))})$ | $\Theta(d^{2-1/(2\alpha)}n^{-4+1/\alpha} \vee d^{-2\alpha}$ $\vee d^{-1/(2\alpha)}n^{-2+1/\alpha})$ |

| | | [VF22][10] (see also [52]) | |
|---|---|---|---|
| **Algorithm** | | one-pass Accelerated SGD ($n \geq d$) | |
| **Risk** $\mathscr{P}(n)$ | Phase Ia | $O(dn^{-2}\|\theta_0 - \theta^\star\|^2 \vee d^{-\rho})$ | |
| | Phase II | $O(dn^{-2}\|\theta_0 - \theta^\star\|^2 \vee d^{-2\alpha})$ | |
| | Phase III | $O(dn^{-2}\|\theta_0 - \theta^\star\|^2 \vee d^{-2\alpha})$ | |

| | | This work [Paq+24][1] | [DLM24][2] |
|---|---|---|---|
| **Algorithm** | | one-pass SGD/SGD-M | RR + $O(1)$-ridge |
| **Risk** $\mathscr{P}(n)$ | Phase Ia | $\Theta(n^{-\rho/(2\alpha)} \vee d^{-\rho})$ | same as [Paq+24][1] |
| | Phase II | $\Theta(n^{-\rho/(2\alpha)} \vee d^{-1}n^{-1+1/(2\alpha)} \vee d^{-2\alpha})$ | same as [Paq+24][1] |
| | Phase III | $\Theta(n^{-2+1/(2\alpha)} \vee d^{-1}n^{-\frac{2\alpha-1}{2\alpha}} \vee d^{-2\alpha})$ | $\Theta(n^{-2} \vee d^{-1}n^{-\frac{2\alpha-1}{2\alpha}} \vee d^{-2\alpha})$ |

| | | Minimax optimal[789] | [Lin+24][6] |
|---|---|---|---|
| **Algorithm** | | one-pass SGD, very small stepsize | one-pass SGD, very small stepsize |
| **Risk** $\mathscr{P}(n)$ | Phase Ia | $O(n^{-\rho/(2\alpha+2\beta)})$ | $\Theta(d^{-\rho} + n^{-\rho/(2\alpha)} + \min\{\frac{d}{n}, n^{-1+1/(2\alpha)}\})$ |
| | Phase II | $O(n^{-\rho/(2\alpha+2\beta)})$ | does not cover |
| | Phase III | $O(n^{-4\alpha/(4\alpha+1)})$ | does not cover |

[7] Carratino, Rudi, Rosasco. *Learning with sgd and random features.* 2018

[8] Dieuleveut and Bach. *Nonparametric stochastic approximation with large stepsizes.* 2016.

[9] Pillaud-Vivien, Rudi, Bach. *Statistical optimality of SGD on hard learning problems through multiple passes.* 2018.

[9] Varre, Pillaud-Vivien, Flammarion. *Last iterate convergence of SGD for least squares in the interpolation regime* 2021.

[10] Varre, Flammarion. *Accelerated SGD for Non-Strongly-Convex Least Squares* 2022.

Table 2: **(Nonexhaustive) Comparison of sample-complexity results.** Let $\rho \overset{\text{def}}{=} 2\alpha + 2\beta - 1$. We use $n =$ sample size($t$ in our notation), $d =$ parameters. [DLM24][1] can also be done with RR+optimal-ridge, which yields same in Phase Ia, but different in Phase II/III. [VPF21][9] obtain $\mathscr{P} \ll n^{-\min\{1/(2\alpha),(2\alpha+2\beta-1)/(2\alpha)\}}$, that is, they capture the $\mathcal{F}_{pp}$, but not $\mathcal{F}_{ac}$. The *minimax* optimal SGD rates never achieve any of the rates (always worse), which can be connected to overly conservative, small stepsizes. For derivation of the minimax rates, we used Cor. 2 from [DB18][7]. [Lin+24][5] requires label noise order 1 and also a very small learning rate. For [VF22][10], we believe (though not proven) after numerical experiments that $\|\theta_0 - \theta^\star\|^2 \lesssim d^{1-2\beta}$ in Phase Ia and otherwise constant in Phase II/III. As it takes on the order of $d^{\alpha+1/2}$ to reach stationarity for DANA-constant, we see that our bounds improve over [VF22][10], but similar. In particular, the two results agree as one approaches stationarity. For all algorithms, we set batch size $= 1$.

| | **Equivalent Hyperparameters in Alg. (Gen-Mom-SGD)** | **Good Choices for Hyperparameters** |
|---|---|---|
| **AcSGD[94]** $(\tilde{\alpha}, \tilde{\beta})$ | $\gamma_1(t) = \frac{\tilde{\alpha}(t+1)^2}{t+2} - \frac{t+1}{t+2}\tilde{\beta},$ $\gamma_2(t) = \tilde{\beta}, \gamma_3(t) = \frac{1}{t+1},$ $\Delta(t) = \frac{1}{t+2}$ | $\tilde{\alpha} = c_1/(d \times \mathrm{Tr}(D)), \tilde{\beta} = c_2/\mathrm{Tr}(D),$ $c_1, c_2$ constants; See [94, Thm 3] |
| **S-Nesterov** $(\tilde{\gamma})$ | $\gamma_1(t) = \frac{\tilde{\gamma}(t+1)^2}{t+2} - \frac{t+1}{t+2}\tilde{\gamma},$ $\gamma_2(t) = \tilde{\gamma}, \gamma_3(t) = \frac{1}{t+1},$ $\Delta(t) = \frac{1}{t+2}$ | $\tilde{\gamma} = c_2/\mathrm{Tr}(D), c_2$ constant; See [40] for details |
| **Schedule-Free SGD** [33] $(\tilde{\gamma}, \tilde{\beta})$ | $\gamma_1(t) = 1, \gamma_2(t) = \tilde{\gamma}(1 - \tilde{\beta}),$ $\gamma_3(t) = \frac{\tilde{\gamma}\tilde{\beta}}{t+1}, \Delta(t) = \frac{1}{t+1}$ | $\tilde{\beta} = 0.9, \tilde{\gamma} = c_2/\mathrm{Tr}(D), c_2$ constant; See [73] for equivalence proof. |
| **ASGD** [61] $(\tilde{\gamma}, \tilde{\alpha}, \tilde{\beta}, \tilde{\delta})$ | $\gamma_1(t) = \tilde{\alpha}(\tilde{\delta} - \tilde{\gamma}),$ $\gamma_2(t) = \tilde{\alpha}\tilde{\delta} + (1 - \tilde{\alpha})\tilde{\gamma},$ $\gamma_3(t) = -(1 - \tilde{\alpha})(1 - \tilde{\beta}),$ $\Delta(t) = (1 - \tilde{\beta})\tilde{\alpha}$ | See [61] for choices of hyperparameters; Time-independent hyperparameters $(\tilde{\gamma}, \tilde{\alpha}, \tilde{\beta})$, but some depend on $1/d$. |
| **SGD with 1-memory** [107] $(q_0, \tilde{\alpha}, \tilde{\delta})$ | $\gamma_1(t) = 1,$ $\gamma_2(t) = q_0,$ $\gamma_3(t) = (1 + t)^{-\tilde{\alpha}},$ $\Delta(t) = (1 + t)^{-\tilde{\delta}}$ | Heuristic only (no specific constants) $0 < \tilde{\alpha} < (2\alpha)^{-1},$ $0 < \tilde{\delta} \leq 1.$ |

Table 3: **Other algorithms in the form of updates given by (Gen-Mom-SGD) and hyperparameters**. See [40, 73, 94] for details. Although not proven, AcSGD [94] should attain similar scaling laws as DANA-constant and ASGD [61] should attain similar scaling laws as SGD. We note that ASGD [61] is not quite in the form of (Gen-Mom-SGD) as the update in $y_t$ uses a stale gradient $\nabla\mathscr{R}(\theta_{t-1}; x_t)$ instead of $\nabla\mathscr{R}(\theta_t; x_{t+1})$ for (Gen-Mom-SGD). SGD with 1-Memory [107] is a heuristic algorithm independently developed at the same time as DANA-decaying and very similar; No proof is given. We believe our result proves this algorithm as well.

The method of analysis of the risk curves in this paper is by formulation of a Volterra equation. For SGD and stochastic momentum, these Volterra equations were studied in the high-dimensional regime, see [60, 77, 78, 79, 80]. Particularly, we use the formulation that derives the Volterra equation via a system of coupled difference equations for weights of the residuals in the observed data covariance. This has been shown to generalizes beyond the least-squares context, at least under SGD, [26]; in isotropic instances, this simplifies to a finite-dimensional family of ODES [4]. This can also be generalized to momentum SGD methods [78] and large batch SGD methods [60]. Convolution-Volterra equations are convenient tools, as they are well-studied parts of renewal theory [5] and branching process theory [8].

Another method of analysis is dynamical mean field theory. The closest existing work to this one in scientific motivations is [16, 17], which uses this technique. This formally can be considered as a type of Gaussian process approximation, but for a finite family of observables ("order parameters"). In instances of one-pass SGD (including in anisotropic cases), this is rigorously shown to hold in [42]. The analysis of the resulting self-consistent equations is nontrivial, and [7, 16, 17] does some of this analysis under simplifying assumptions on the structure of the solutions of these equations.

Besides these works, there is a large theory around generalization error of SGD. The work of [95] gives a direct analysis of risks of SGD under "source/capacity" type assumptions which formally capture the $F_{pp}$ parts of the Phase Ia/II loss curves. The risk bounds of [111] give non-asymptotic estimates which again reproduce tight estimates for the $F_{pp}$ parts of the loss (note that to apply these bounds to this case, substantial random matrix theory needs to be worked out first); see also [63] where some of this is done.

# B  Additional Algorithm Set-up

Let $\gamma_1, \gamma_2, \gamma_3 : [0, \infty) \to (0, \infty)$ be learning rate schedules and $\Delta : [0, \infty) \to [0, \infty)$ be the momentum schedule. To solve the PLRF (3), we use a class of *one-pass* (mini-batch) stochastic momentum algorithms with batch size $B$. Let $y_{-1} = \theta_0 \in \mathbb{R}^d$. At each iteration $t \geq 0$, we generate independent, new samples $\{x_{t+1}^i\}_{i=1}^B$ and update by

$$y_t = (1 - \Delta(t))y_{t-1} + \gamma_1(t) \times \sum_{i=1}^B W^T x_{t+1}^i \big(\langle W^T x_{t+1}^i, \theta_t \rangle - \langle x_{t+1}^i, b \rangle\big)$$

$$\theta_{t+1} = \theta_t - \gamma_2(t) \times \sum_{i=1}^B W^T x_{t+1}^i \big(\langle W^T x_{t+1}^i, \theta_{t+1}^i \rangle - \langle x_{t+1}^i, b \rangle\big) - \gamma_3(t) \times y_t, \tag{13}$$

where $\gamma_i(t), \Delta(t)$ are non-negative functions. We will consider multiple versions of the algorithm (13), that is, with different choices for the learning rates $\gamma$'s and the momentum schedule $\Delta(t)$. In Section 3, we summarize the different algorithms we consider based on hyperparameters as well as good choices for those hyperparameters.

## B.1  Stochastic gradient descent (SGD)

An example of a known stochastic algorithm that falls within the update rule given in (13) is the *stochastic gradient descent (SGD) with learning rate schedule, $\gamma_2(t)$*. This algorithm is determined by setting $\gamma_1, \gamma_3 \equiv 0$, $\Delta \equiv 1$, and $\gamma_2(t) : [0, \infty) \to (0, \infty)$ a learning rate schedule in (13). Specifically, it updates by

$$\theta_{t+1} = \theta_t - \gamma_2(t) \times \sum_{i=1}^B W^T x_{t+1}^i \big(\langle W^T x_{t+1}^i, \theta_{t+1}^i \rangle - \langle x_{t+1}^i, b \rangle\big). \tag{14}$$

In Section F, we derive the deterministic ODEs and analyze SGD for compute-optimality when the learning rate schedule $\gamma_2(t)$ is a decreasing function.

### B.1.1  SGD with constant learning rate

The compute-optimal curves for SGD when $\gamma_2(t)$ is a constant was studied extensively in [16, 63, 80]. In [80], a necessary and sufficient condition for convergence of the algorithm based on the learning rate $\gamma_2$ was established (see [80, Prop. C.2]). Particularly, the condition was

$$\gamma_2 < \frac{2}{B+1} \quad \text{and} \quad \int_0^\infty \mathcal{K}(t) \, dt < 1,$$

where $\mathcal{K}$ is the kernel function for SGD (see Section F for a specific definition). Asymptotically, we know that $\mathcal{K}$ ([80, Cor. G.1]) satisfies

$$\int_0^\infty \mathcal{K}(t) \, dt \sim \frac{\gamma_2}{2} \sum_{j=1}^v j^{-2\alpha}.$$

Therefore, we have two cases to consider: $2\alpha < 1$ (below high-dimensional line) and $2\alpha > 1$ (above high-dimensional line).

**Remark B.1** (Stability conditions for SGD with constant learning rate). *Let $Tr(D) \stackrel{def}{=} \sum_{j=1}^v j^{-2\alpha}$. When $2\alpha < 1$, suppose that $\frac{v}{d} \to r \in (1, \infty)$. The stability conditions for SGD with constant learning rate $\gamma_2$ are*

$$
\begin{aligned}
(2\alpha > 1): & \qquad \gamma_2 < \tfrac{2}{B+1} \quad \text{and} \quad \gamma_2 < \tfrac{2}{Tr(D)} \\
(2\alpha < 1): & \quad \gamma_2 < \tfrac{2}{B+1} \quad \text{and} \quad \gamma_2 < \tfrac{2}{Tr(D)} \sim \tfrac{2(1-2\alpha)}{d^{(1-2\alpha)}}.
\end{aligned}
$$

*This means that for $2\alpha > 1$, the learning rate is always constant (order 1) as the $Tr(D)$ is summable. Below the high-dimensional line, $Tr(D)$ grows with the number of features $d$.*

## B.2 Classic (constant) momentum (SGD-M)

In this section, we consider the classical stochastic momentum where the learning rates $\gamma_1 = 1, \gamma_2 = \tilde{\gamma}_2 d^{-\kappa_1}$, and $\gamma_3 = \tilde{\gamma}_3 d^{-\kappa_2}$, and momentum, $\Delta \equiv \delta$, is constant (does not change with time and order 1 with respect to $d$). We call this algorithm *stochastic gradient descent with momentum (SGD-M)*. In particular, we note that (stochastic) heavyball [82] occurs when $\gamma_2 = 0$ and $\gamma_3$ and $\Delta$ are specific constants determined by the largest and smallest (non-zero) eigenvalue of $\check{K}$. In this case, the updates in (13) with $y_{-1} = \theta_0 = 0$ follow

$$
y_t = (1 - \delta)y_{t-1} + \sum_{i=1}^{B} W^T x_{t+1}^i (\langle W^T x_{t+1}^i, \theta_t \rangle - \langle x_{t+1}^i, b \rangle)
$$

$$
\theta_{t+1} = \theta_t - \gamma_2 \sum_{i=1}^{B} W^T x_{t+1}^i (\langle W^T x_{t+1}^i, \theta_t \rangle - \langle x_{t+1}^i, b \rangle) - \gamma_3 \times y_t.
$$

(15)

We will show a sufficient condition on the hyperparameters $\gamma_2, \gamma_3, \delta$ in Cor. G.1 for the simplified Volterra equation (55) to remain bounded. However, we emphasis that there are discrepancies between this stability condition and the *actual* stability condition of the algorithm. Indeed, the simplified Volterra equation neglects $\Delta^2$ terms which makes any $\Delta > 0$ converge. On the other hand, it is possible to make a similar analysis on a different set of ODEs (coin-flip ODEs (49)) with $\gamma_2 = 0$. In that case, we can derive an explicit condition for stability of the algorithm. Specifically, the stability condition takes the form $\frac{2\delta(2-\delta)}{2(2B+1)+\delta(3B+1)} > \gamma_3 > 0$ with $\delta \in (0, 2)$. The condition $\delta \in (0, 2)$ is crucial to avoid exponential growth of the sequence $(y_t, t \geq 0)$. This shows a limitation of the simplified ODEs (43) which may miss some stability conditions.

## B.3 Dimension-Adapted Nesterov Acceleration: DANA-constant

Another important algorithm we consider is DANA, introduced in [78], where it was shown to accelerate SGD in the proportional $d$ and sample setting (a.k.a. thermodynamic limit). The main distinction with (DANA) over SGD-M is the momentum schedule. Following a Nesterov style momentum, we set $\Delta(t) = \frac{\delta}{1+t}$. In DANA-constant, the learning rates are all set to be constant (possibly dimension-dependent). We will consider a more general setting for DANA-constant in the appendix then what was introduced in the main introduction. Setting $y_{-1} = \theta_0 = 0$, the DANA-constant algorithm updates as

(DANA-constant)
$$
y_t = (1 - \Delta(t))y_{t-1} + \gamma_1 \times \sum_{i=1}^{B} W^T x_{t+1}^i (\langle W^T x_{t+1}^i, \theta_t \rangle - \langle x_{t+1}^i, b \rangle)
$$

$$
\theta_{t+1} = \theta_t - \gamma_2(d) \times \sum_{i=1}^{B} W^T x_{t+1}^i (\langle W^T x_{t+1}^i, \theta_t \rangle - \langle x_{t+1}^i, b \rangle) - \gamma_3(d) \times y_t.
$$

$$
\text{where } \Delta(t) = \frac{\delta}{t+1}, \quad \gamma_1 = 1, \quad \gamma_2(d) = \frac{\text{constant}}{d^{\kappa_1}}, \quad \text{and} \quad \gamma_3(d) = \frac{\text{constant}}{d^{\kappa_2}}.
$$

(16)

Here we have batch size $B = d^{\kappa_b}$ where the exponents $\kappa_b, \kappa_1, \kappa_2 \geq 0$. For the DANA-constant introduced in the main introduction, we set $\kappa_2 = \kappa_1 + 1$ and $\kappa_b = 0$. At times throughout the appendix, we will reduce back down to DANA-constant with these specific parameters, but for the proof, see Section H, we will use this more general DANA-constant setting.

**Remark B.2** (Good choices for hyperparameters of DANA-constant.)**.** *In Lemma H.5, we give sufficient conditions for stability of the solution to the Volterra equation* (55) *on the simplified system of ODEs* (43) *that accurately predicts the dynamics of the stochastic algorithm DANA-constant. For necessary conditions on the hyperparameters (on the simplified ODEs), see Corollary H.2. Some good choices for hyperparameters of DANA-constant (in batch size $B = 1$) are*

$$
\delta > 4 \times \max\left\{ \frac{2\alpha + 2\beta - 1}{\alpha}, 4 - \frac{1}{\alpha} \right\}, \quad \gamma_1 = 1, \quad \gamma_2 = \frac{c_2}{Tr(D)}, \quad \text{and} \quad \gamma_3 = \frac{c_3}{d \times Tr(D)},
$$

$$
\text{satisfying} \quad \frac{\gamma_1 \gamma_3}{2\gamma_2} d + \frac{\gamma_2}{2} Tr(D) < 1.
$$

*Here $c_2, c_3$ are positive constants and the matrix $D = Diag(j^{-2\alpha} : 1 \leq j \leq v)$.*

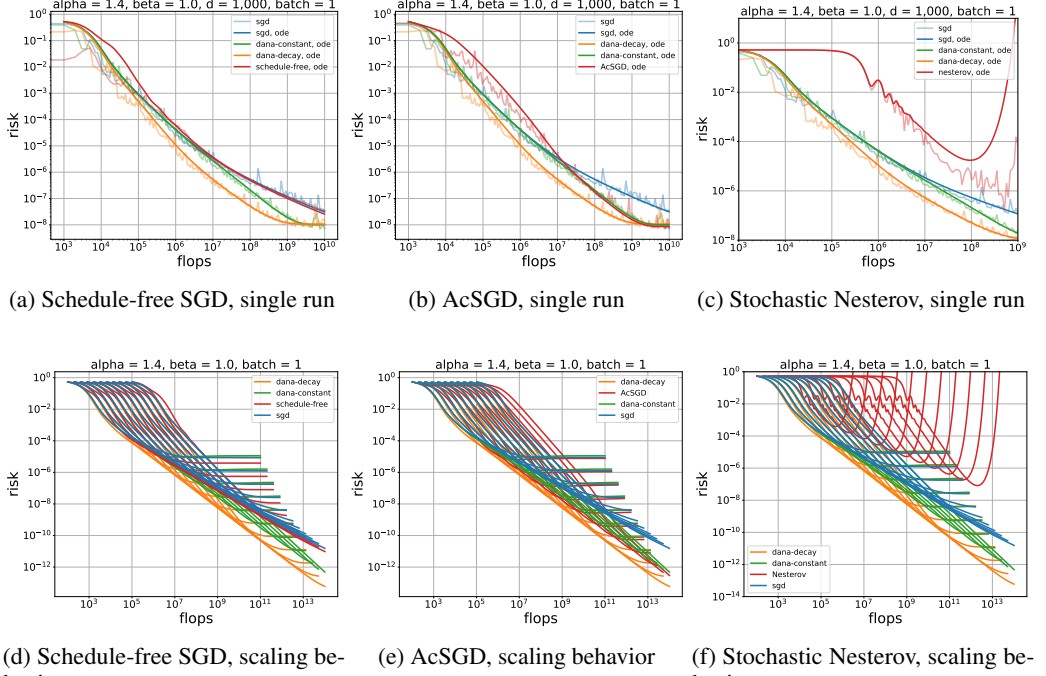

(a) Schedule-free SGD, single run

(b) AcSGD, single run

(c) Stochastic Nesterov, single run

(d) Schedule-free SGD, scaling behavior

(e) AcSGD, scaling behavior

(f) Stochastic Nesterov, scaling behavior

Figure 10: **Comparison of SGD, DANA with other known algorithms (Schedule-free, AcSGD, Nesterov).** Numerical setup: SGD (blue curves) learning rate $\gamma_2 = 0.5/\text{Tr}(D)$, DANA-constant (green) has $\gamma_1(t) = 1$, $\gamma_2(t) = 0.5/\text{Tr}(D)$, $\gamma_3(t) = 0.1/d$, $\Delta(t) = \delta/(1+t)$ where $\delta = \max\{2 - 1/\alpha, (2\alpha + 2\beta - 1)/\alpha\} + 1$; DANA-decaying (orange) same $\gamma_1, \gamma_2$, and $\Delta$ as DANA-constant, $\gamma_3(t) = 0.1/(1+t)^{1/(2\alpha)}$; Schedule-free SGD (red in 1st column) $\tilde{\beta} = 0.9$ and $\tilde{\gamma} = 0.5/\text{Tr}(D)$ in [33]; AcSGD (red in 2nd column) $\tilde{\alpha} = 0.1/\text{Tr}(D) \times 1.0/\text{Tr}(D)$ and $\tilde{\beta} = 0.4/\text{Tr}(D)$ as defined in [94]; Stochastic Nesterov (red in 3rd column) $\tilde{\alpha} = \tilde{\beta} = 0.1/\text{Tr}(D) \times 1.0/d^{1/2}$ as defined in [94]. Top row: algorithms were run for $10^7$ steps with $d = 1000$, $v = 10000$; Bottom row: algorithms run using the ODEs with equivalences given in Table 3; $10^9$ iterations of algorithm, $d = \{100 \times 2^i\}$, $i = 1, \ldots, 10$ and $v = 10 \times d$. **(1st column: Schedule-free SGD [33])** Schedule-free SGD (red) scales very closely with SGD (blue) for this $(\alpha, \beta)$. We see both DANA-decay and DANA-constant accelerate. **(2nd column: AcSGD [94])** AcSGD (red) gives scaling laws similar to DANA-constant (green). Moreover, it has the same property of changing behavior at $t \asymp d$. Although an upper bound was proven for AcSGD in [94], this is not optimal when applied to the PLRF and, in particular, one would need to understand $\|\theta_0 - \theta^\star\|^2$. A similar technique as our approach to DANA-constant should apply to AcSGD. **(3rd column: stochastic Nesterov)** Stochastic Nesterov is known to not converge (see e.g., Thm.7 [38] and references therein).

## B.4 Data-Adapted Nesterov Acceleration (DANA-decaying): Time/Data-dependent learning rates

In the same spirit as (16), we consider a variation of DANA-constant, called DANA-decaying. In this case, we not only have a momentum time-dependent schedule, but we also allow the learning rates to be time dependent. The motivation for a decaying $\gamma_3$ schedule is, in light of Section 3, to avoid the trivial behavior of DANA-constant for small $t$ by using larger $\gamma_3$ at the start, while decaying it progressively to reduce the noise and preserve the stability of the algorithm. Therefore, we introduce the DANA-decaying algorithm given by

$$(\text{DANA-decaying}) \quad \begin{aligned} y_t &= (1 - \Delta(t))y_{t-1} + \gamma_1 \times \sum_{i=1}^B W^T x_{t+1}^i(\langle W^T x_{t+1}^i, \theta_t \rangle - \langle x_{t+1}^i, b \rangle) \\ \theta_{t+1} &= \theta_t - \gamma_2(d) \sum_{i=1}^B W^T x_{t+1}^i(\langle W^T x_{t+1}^i, \theta_t \rangle - \langle x_{t+1}^i, b \rangle) - \gamma_3(t; d)y_t. \end{aligned}$$

$$\text{where } \Delta(t) = \frac{\delta}{t+1}, \ \gamma_3(t; d) = \frac{\text{constant}}{(1+t)^{\kappa_3}}, \ \text{ and } \ \gamma_2(d), \gamma_1 = \text{constants}.$$

(17)

Here the exponent $\kappa_3 \geq 0$ and batch size $B$ is independent of $d$. We only prove the result when $2\alpha > 1$ and $\gamma_2(d) \asymp 1$, see Section I. In the setting of $2\alpha > 1$, this amounts to choosing (up to an absolute constant) the maximal learning rate. We provide some good choices for the hyperparameters that work well across scales on the PLRF.

**Remark B.3** (Good hyperparameters for DANA-decaying, with $B = 1$). *In Section I, we provide some heuristics as to the correct sufficient conditions for stability of DANA-decay. More precisely for any $1 \geq \kappa_3 > \frac{1}{2\alpha}$ (for DANA-decaying, one should pick $\kappa_3 = \frac{1}{2\alpha}$) the conditions are stated as*

$$\delta + \frac{\kappa_3}{2} > \left(1 - \frac{\kappa_3}{2}\right) \max\left\{\frac{2\alpha + 2\beta - 1}{\alpha}, 4 - \frac{1}{\alpha}\right\}, \quad \gamma_1 = 1, \quad \gamma_2(d) = \frac{c_2}{Tr(D)},$$

$$\text{and} \quad \gamma_3(t; d) = \frac{c\gamma_2(d)}{(1+t)^{\kappa_3}}.$$

*Here the constants $c_2, c > 0$ should be chosen small enough and the matrix $D = Diag(j^{-2\alpha} : 1 \leq j \leq v)$. Below the high-dimensional line $(2\alpha < 1)$, we are not certain if another choice of $\gamma_3$ may accelerate.*

In the rest of the appendix, we use the following.

> **Remark B.4** (DANA-decaying/DANA-constant.). *Unless it is clear that $\kappa_2/\kappa_3$ are being used, e.g., in the proofs of DANA-constant/DANA-decaying (Section H and Section I), DANA-constant refers to $(\kappa_2 = 1 + \kappa_1, \kappa_3 = 0)$ and DANA-decaying refers to $(\kappa_2 = 0, \kappa_3 = 1/(2\alpha))$. In some cases, we will further reduce to the setting where batch size is order 1 in d. We note when we refer to DANA-decaying we **always** assume that batch size is order 1.*

## B.5 Comparison with Schedule-free, AcSGD, Stochastic Nesterov and conjectured scaling laws

Other algorithms (e.g., stochastic Nesterov, AcSGD [94], Accelerated SGD [52, 61]) can be written in terms of Gen-Mom-SGD. For the equivalence of hyperparameters of these algorithms to the hyperparameters in Gen-Mom-SGD, see Table 3. See also Figure 10 for their scaling performance compared with DANA and SGD.

We conjecture that Theorem 4.1 can be extended to include a time-dependent $\gamma_1(t; d)$. In some sense, the algorithm Gen-Mom-SGD is over-parameterized, that is, for Gen-Mom-SGD$(\gamma_1, \gamma_2, \gamma_3, \Delta \asymp (1+t)^{-1})$ such that

$$\gamma_1(t; d) \times \gamma_3(t; d) = \hat{\gamma}_3(t; d) \quad \Rightarrow \quad \text{Gen-Mom-SGD}(\gamma_1, \gamma_2, \gamma_3, \Delta \asymp (1+t)^{-1}) \approx \text{DANA}(\gamma_2, \hat{\gamma}_3).$$

This is certainly true when $\gamma_1$ is a constant (d-dependent allowed). It should also hold (at least asymptotically) when $\gamma_1(t; d)$ and $\gamma_3(t; d)$ are time-dependent provided that $\gamma_1(t; d)$ is growing and $\Delta(t)$ is nice. Intuitively, this is true as $\theta_{t+1}$ in Gen-Mom-SGD gets updated by approximately $\gamma_1(t; d) \times \gamma_3(t; d) \times y_t$ after unrolling the recursion on $y_t$'s.

Under this belief and using the equivalence with our parametrization in Table 3 we can very precisely conjecture their scaling law behavior.

- **Schedule-free SGD.** In our parametrization, Schedule-free SGD uses the asymptotic learning rates $\gamma_1(t;d)\gamma_3(t;d) = \frac{\tilde{\gamma}\tilde{\beta}}{1+t}$ and $\gamma_2(t;d) = \tilde{\gamma}(1-\tilde{\beta})$ with $\tilde{\gamma} \asymp \frac{1}{\text{Tr}(D)}$, $\tilde{\beta} \asymp 1$. Therefore, we see Schedule-free SGD is approximately DANA-decaying with $\gamma_2(t;d) \asymp \frac{1}{\text{Tr}(D)}$ and $\gamma_3(t;d) = \frac{1}{\text{Tr}(D)(1+t)}$ or, if you like, $\kappa_3 = 1$. Then from Theorem 4.1, we have that the function $\vartheta(t) \asymp 1 + 2\gamma_2 B t$. This leads to the conjectured behavior:

> **Conjectured Scaling Laws for Schedule-free SGD:** For any $\alpha > 0$, $2\alpha + 2\beta > 1$, Schedule-free SGD has the exact same scaling laws as SGD.

- **AcSGD.** In our parametrization, AcSGD [94] uses the asymptotic learning rates $\gamma_1(t)\gamma_3(t) = \tilde{\alpha}$ and $\gamma_2(t) = \tilde{\beta}$ with $\tilde{\alpha} \asymp \frac{1}{d \times \text{Tr}(D)}$ and $\tilde{\beta} \asymp \frac{1}{\text{Tr}(D)}$. We see that this exactly corresponds for large time to the schedule for DANA-constant with $\gamma_3(t;d) = \gamma_2(d) \times \frac{1}{d}$. Using our result on DANA-constant Theorem H.3, this leads to the conjectured behavior:

> **Conjectured Scaling Laws for AcSGD:** For any $\alpha > 0$, $2\alpha + 2\beta > 1$, AcSGD has the exact same scaling laws as DANA-constant.

- **Stochastic Nesterov** In our parametrization, stochastic Nesterov uses the asymptotic learning rates $\gamma_1(t)\gamma_3(t) = \tilde{\gamma}$ and $\gamma_2(t) = \tilde{\gamma}$ with $\tilde{\gamma} \asymp \frac{1}{\text{Tr}(D)}$. We see that this exactly corresponds for large time to the schedule for DANA-constant without the $\frac{1}{d}$ scaling in the $\gamma_3(t)$ learning rate. Using our stability result on DANA-constant Corollary H.2, this leads to the conjectured behavior (see related results in e.g., Thm.7 [38] and references therein):

> **Conjectured Scaling Laws for Stochastic Nesterov:** For any $\alpha > 0$, $2\alpha + 2\beta > 1$, Nesterov Momentum diverges for large times.

## C   Deriving ODEs and Volterra equations

In this section, we derive Volterra equations that match the dynamics of the meta stochastic momentum algorithm updates in (13). We develop two systems of ODEs that culminate in two (related) Volterra equations. In Section C.1, we deduce a system of ODEs and a Volterra equation that, in expectation, matches the expected loss under the updates (13). We denote these as the *exact ODEs* and the *exact Volterra equation*, resp. While numerically solvable, analyzing the *exact ODEs* presents significant challenges. Therefore, we introduce a system of SDEs derived from studying stochastic momentum algorithms in the high-dimensional optimization framework [78]. In Section C.2, we derive the ODEs and Volterra equation for the SDE framework (*simplified ODEs*), resp. We empirically demonstrate that this alternative simplified ODE system accurately model the behavior of SGD/SGD-M, DANA-constant, and DANA-decaying (see Fig. 26, 27, 28, 29, 30, 31, 32, 33), and are considerably simpler for analyzing scaling laws. Throughout our analysis, we clearly indicate which system of ODEs/Volterra equation we are examining for each algorithm.

### C.1   Derivation of exact ODEs and exact Volterra for the expected loss of meta stochastic momentum algorithm, (Gen-Mom-SGD)

In this section, we derive a Volterra equation for the dynamics of the meta stochastic momentum algorithm which updates using (13). Let $N_t$ be an iid Poisson process. At a jump of $N_t$, we generate $\{x^i\}_{i=1}^B$ new data points whose coordinates follow an $\alpha$-power law and update by

$$Y_t = (1 - \Delta(t))Y_{t-} + \gamma_1(t) \times \sum_{i=1}^B W^T x_{N_t}^i (x_{N_t}^i)^T (W\Theta_{t-} - b) \stackrel{\text{def}}{=} Y_{t-} + \Delta_t^y(Y_{t-}, \Theta_{t-})$$

$$\Theta_t = \Theta_{t-} - \gamma_2(t) \times \sum_{i=1}^B W^T x_{N_t}^i (x_{N_t}^i)^T (W\Theta_{t-} - b) - \gamma_3(t) \times Y_t \stackrel{\text{def}}{=} \Theta_{t-} + \Delta_t^\theta(Y_t, \Theta_{t-}).$$

$$(18)$$

Here we naturally extend time $t$ to be a continuous parameter with $Y_{t-} = Y_{t-1}$ and we start this process so that $\Theta_0 = \theta_0$ and $Y_{0-} = y_{-1}$, that is, at initialization the two stochastic algorithms agree. This is a standard way to embed discrete processes into a continuous process.

For any $(\Theta, Y) \mapsto f(\Theta, Y) \in \mathbb{R}$, the mean behavior is given by

$$\frac{\mathrm{d}}{\mathrm{d}t} \mathbb{E}\left[f(Y_t, \Theta_t)\right] = \mathbb{E}\left[f(Y_t + \Delta_t^y(Y_t, \Theta_t), \Theta_t + \Delta_t^\theta(Y_t, \Theta_t)) - f(Y_t, \Theta_t)\right]. \tag{19}$$

### C.1.1 Derivation of the Exact ODEs.

For this, we need to introduce statistics that are closed under differentiation as defined in (19).

To do so, we begin by writing $\mathbb{R}^v = \mathrm{Im}(W) \oplus W^\perp$. Thus, there exists a $\check{b} \in \mathbb{R}^d$ and $\dot{b} \in \mathbb{R}^v$ such that one can write $b = W\check{b} + \dot{b}$, that is, we can decompose $b$ as an element in the image of $W$ and an element in the co-ker of $W$. Formally, we have that

$$b = W\check{b} + \dot{b}, \quad \text{where } W^T D \dot{b} = 0.$$

We will now choose some specific functions/statistics that will close under differentiation. Let $\check{K} \stackrel{\text{def}}{=} W^T D W \in \mathbb{R}^{d \times d}$ and let $(\lambda_j, \omega_j)_{j=1}^d$ by the eigenvalue-eigenvector pairs for $\check{K}$. We will consider the follow special statistics/functions for each eigenvector of $\check{K}$

$$\rho_j^2(t) = \mathbb{E}\left[\langle \omega_j^{\otimes 2}, (\Theta_t - \check{b})^{\otimes 2}\rangle | W\right], \qquad \xi_j^2(t) = \mathbb{E}\left[\langle \omega_j^{\otimes 2}, Y_t^{\otimes 2}\rangle | W\right],$$
$$\text{and} \quad \chi_j(t) = \mathbb{E}\left[\langle \omega_j^{\otimes 2}, (\Theta_t - \check{b}) \otimes Y_t\rangle | W\right]. \tag{20}$$

We recover the expected loss function from the statistics (20) by

$$\mathbb{E}\left[\mathscr{P}(\Theta_t)|W\right] = \mathbb{E}\left[\mathscr{P}(\Theta_\infty)|W\right] + \sum_j \lambda_j \rho_j^2(t), \quad \text{where } \mathbb{E}\left[\mathscr{P}(\Theta_\infty)|W\right] = \lim_{t \to \infty} \mathbb{E}\left[\mathscr{P}(\Theta_t)|W\right].$$

Therefore, knowing the evolution of $\rho_j^2(t)$ and $\mathbb{E}\left[\mathscr{P}(\Theta_\infty)|W\right]$ for every $j$ allows us to recover information about the evolution of the loss curve under $\Theta_t$. Moreover, we will show that the functions $(\rho_j^2, \xi_j^2, \chi_j^2)$ close, that is, there is a closed system of ODEs that defined their behavior.

First, we observe that we will need moments of Gaussians via Wick's formula. In particular, for fixed vectors $v_i \in \mathbb{R}^v$, $i = 1, 2, 3, 4$ and $x_j = j^{-\alpha} z_j$ where $z_j \sim N(0, 1)$,

$$\mathbb{E}_x[x\langle x, v_1\rangle] = Dv_1$$
$$\mathbb{E}_x[\langle x, v_1\rangle\langle x, v_2\rangle\langle x, v_3\rangle\langle x, v_4\rangle] = \langle D, v_1 \otimes v_2\rangle\langle D, v_3 \otimes v_4\rangle + \langle D, v_1 \otimes v_3\rangle\langle D, v_2 \otimes v_4\rangle$$
$$+ \langle D, v_1 \otimes v_4\rangle\langle D, v_2 \otimes v_3\rangle.$$

The $(v \times v)$-matrix $D \stackrel{\text{def}}{=} \mathrm{Diag}(j^{-2\alpha} : 1 \leq j \leq v)$ is the covariance of $x$. Using these moment computations, we can compute $\frac{\mathrm{d}}{\mathrm{d}t}$ for the functions in (20).

$\underline{\rho_j^2 \text{ functions:}}$  Using (19) applied to $\mathbb{E}\left[f(\Theta_t, Y_t)\right] = \rho_j^2(t)$ yields,

$$\begin{aligned}
\frac{\mathrm{d}}{\mathrm{d}t}\rho_j^2(t) = {}& 2\mathbb{E}\left[\langle \omega_j^{\otimes 2}, \left(-\gamma_2(t)\sum_{i=1}^B W^T x_t^i (x_t^i)^T (W\Theta_t - b)\right) \otimes (\Theta_t - \check{b})\rangle | W\right] \\
& + 2\mathbb{E}\left[\langle \omega_j^{\otimes 2}, -\gamma_3(t)(1 - \Delta(t))Y_t \otimes (\Theta_t - \check{b})\rangle | W\right] \\
& + 2\mathbb{E}\left[\langle \omega_j^{\otimes 2}, \left(-\gamma_1(t)\gamma_3(t)\sum_{i=1}^B W^T x_t^i (x_t^i)^T (W\Theta_t - b)\right) \otimes (\Theta_t - \check{b})\rangle | W\right] \\
& + 2\mathbb{E}\left[\langle \omega_j^{\otimes 2}, \left(-\gamma_2(t)\sum_{i=1}^B W^T x_t^i (x_t^i)^T (W\Theta_t - b)\right) \otimes -\gamma_3(t)(1 - \Delta(t))Y_t\rangle | W\right] \\
& + 2\mathbb{E}\left[\langle \omega_j^{\otimes 2}, \left(-\gamma_2(t)\sum_{i=1}^B W^T x_t^i (x_t^i)^T (W\Theta_t - b)\right)\right. \\
& \qquad\qquad \left. \otimes \left(-\gamma_1(t)\gamma_3(t)\sum_{i=1}^B W^T x_t^i (x_t^i)^T (W\Theta_t - b)\right)\rangle | W\right] \\
& + 2\mathbb{E}\left[\langle \omega_j^{\otimes 2}, -\gamma_3(t)(1 - \Delta(t))Y_t \otimes \left(-\gamma_1(t)\gamma_3(t)\sum_{i=1}^B W^T x_t^i (x_t^i)^T (W\Theta_t - b)\right)\rangle | W\right] \\
& + \mathbb{E}\left[\langle \omega_j^{\otimes 2}, \left(-\gamma_2(t)\sum_{i=1}^B W^T x_t^i (x_t^i)^T (W\Theta_t - b)\right)^{\otimes 2}\rangle | W\right] \\
& + \mathbb{E}\left[\langle \omega_j^{\otimes 2}, \left(-\gamma_3(t)(1 - \Delta(t))Y_t\right)^{\otimes 2}\rangle | W\right] \\
& + \mathbb{E}\left[\langle \omega_j^{\otimes 2}, \left(-\gamma_1(t)\gamma_3(t)\sum_{i=1}^B W^T x_t^i (x_t^i)^T (W\Theta_t - b)\right)^{\otimes 2}\rangle | W\right]
\end{aligned}$$

For convenience, we drop the $t$'s on the learning rates $\gamma_i(t) = \gamma_i$ and $\Delta(t) = \Delta$. Applying Wick's rule to the RHS, we get

$$\frac{\mathrm{d}}{\mathrm{d}t}\rho_j^2(t) = -2\gamma_2 B\lambda_j \rho_j^2(t) - 2\gamma_3(1-\Delta)\chi_j(t) - 2\gamma_1\gamma_3 B\lambda_j \rho_j^2(t)$$
$$+ 2\gamma_2\gamma_3(1-\Delta)B\lambda_j\chi_j + 4\gamma_1\gamma_2\gamma_3 B\lambda_j^2\rho_j^2(t) + 2\gamma_1\gamma_2\gamma_3 B\lambda_j \mathbb{E}\left[\|D^{1/2}(W\Theta_t - b)\|^2 | W\right]$$
$$+ 2\gamma_1\gamma_2\gamma_3 B(B-1)\lambda_j^2\rho_j^2(t) + 2\gamma_3^2\gamma_1(1-\Delta)B\lambda_j\chi_j(t) + \gamma_2^2 B\lambda_j \mathbb{E}\left[\|D^{1/2}(W\Theta_t - b)\|^2 | W\right]$$
$$+ 2\gamma_2^2 B\lambda_j^2\rho_j^2(t) + \gamma_2^2 B(B-1)\lambda_j^2\rho_j^2(t) + \gamma_3^2(1-\Delta)^2\xi_j^2(t)$$
$$+ \gamma_1^2\gamma_3^2 B\lambda_j \mathbb{E}\left[\|D^{1/2}(W\Theta_t - b)\|^2 | W\right] + 2\gamma_1^2\gamma_3^2 B\lambda_j^2\rho_j^2(t) + \gamma_1^2\gamma_3^2 B(B-1)\lambda_j^2\rho_j^2(t).$$

$\underline{\xi_j^2 \text{ functions:}}$ Using (19) applied to $\mathbb{E}\left[f(\Theta_t, Y_t)\right] = \xi_j^2(t)$ yields,

$$\frac{\mathrm{d}}{\mathrm{d}t}\xi_j^2(t) = \mathbb{E}\left[\langle \omega_j^{\otimes 2}, \left((1-\Delta(t))Y_t + \gamma_1(t)\sum_{i=1}^{B}W^T x_t^i(x_t^i)^T(W\Theta_t - b)\right)^{\otimes 2}\rangle | W\right]$$
$$- \mathbb{E}\left[\langle \omega_j^{\otimes 2}, Y_t^{\otimes 2}\rangle | W\right]$$
$$= 2\mathbb{E}\left[\langle \omega_j^{\otimes 2}, Y_t \otimes -\Delta(t)Y_t\rangle | W\right] + 2B\mathbb{E}\left[\langle \omega_j^{\otimes 2}, Y_t \otimes \gamma_1(t)\left(W^T x_t x_t^T(W\Theta_t - b)\right)\rangle | W\right]$$
$$+ \mathbb{E}\left[\langle \omega_j^{\otimes 2}, \left(\Delta(t)Y_t\right)^{\otimes 2}\rangle | W\right]$$
$$- 2B\mathbb{E}\left[\langle \omega_j^{\otimes 2}, \Delta(t)Y_t \otimes \gamma_1(t)\left(W^T x_t x_t^T(W\Theta_t - b)\right)\rangle | W\right]$$
$$+ \gamma_1^2(t)B\mathbb{E}\left[\langle \omega_j^{\otimes 2}, \left(W^T x_t x_t^T(W\Theta_t - b)\right)^{\otimes 2}\rangle | W\right]$$
$$+ \gamma_1^2(t)B(B-1)\mathbb{E}\left[\langle \omega_j^{\otimes 2}, \left(W^T x_t^1(x_t^1)^T(W\Theta_t - b)\right) \otimes \left(W^T x_t^2(x_t^2)^T(W\Theta_t - b)\right)\rangle | W\right].$$

Applying Wick's rule, we get

$$\frac{\mathrm{d}}{\mathrm{d}t}\xi_j^2(t) = -2\Delta\xi_j^2(t) + 2\gamma_1 B\lambda_j\chi_j(t) + \Delta^2\xi_j^2(t) - 2\Delta\gamma_1(t)B\lambda_j\chi_j(t)$$
$$+ 2\gamma_1^2 B\lambda_j^2\rho_j^2(t) + \gamma_1^2 B\lambda_j \mathbb{E}\left[\|D^{1/2}(W\Theta_t - b)\|^2 | W\right] + B(B-1)\gamma_1^2(t)\lambda_j^2\rho_j^2(t).$$

$\underline{\chi_j \text{ functions:}}$ Lastly, using (19) applied to $\mathbb{E}\left[f(\Theta_t, Y_t)\right] = \chi_j$ yields,

$$\frac{\mathrm{d}}{\mathrm{d}t}\mathbb{E}\left[\chi_j | W\right] = \mathbb{E}\left[\langle \omega_j^{\otimes 2}, \Theta_t - \gamma_2\sum_{i=1}^{B}W^T x_t^i(x_t^i)^T(W\Theta_t - b)\right.$$
$$- \gamma_3\left((1-\Delta)Y_t + \gamma_1\sum_{i=1}^{B}W^T x_t^i(x_t^i)^T(W\Theta_t - b)\right) - \check{b}$$
$$\otimes \left((1-\Delta)Y_t + \gamma_1\sum_{i=1}^{B}W^T x_t^i(x_t^i)^T(W\Theta_t - b)\right)\rangle | W\right]$$
$$- \mathbb{E}\left[\langle \omega_j^{\otimes 2}, (\Theta_t - \check{b}) \otimes Y_t\rangle | W\right]$$

Applying Wick's rule, we get the following

$$\frac{\mathrm{d}}{\mathrm{d}t}\mathbb{E}\left[\chi_j | W\right] = -\gamma_2 B\lambda_j\chi_j(t) - \gamma_3(1-\Delta)\xi_j^2(t) - \gamma_1\gamma_3 B\lambda_j\chi_j(t) - \Delta\chi_j$$
$$+ \gamma_2\Delta B\lambda_j\chi_j(t) + \gamma_3\Delta(1-\Delta)\xi_j^2(t) + \gamma_1\gamma_3\Delta B\lambda_j\chi_j(t) + \gamma_1 B\lambda_j\rho_j^2(t)$$
$$- 2\gamma_1\gamma_2 B\lambda_j^2\rho_j^2(t) - \gamma_1\gamma_2 B\lambda_j \mathbb{E}\left[\|D^{1/2}(W\Theta_t - b)\|^2 | W\right]$$
$$- \gamma_1\gamma_2 B(B-1)\lambda_j^2\rho_j^2(t) - \gamma_1\gamma_3(1-\Delta)B\lambda_j\chi_j(t) - 2\gamma_1^2\gamma_3 B\lambda_j^2\rho_j^2$$
$$- \gamma_1^2\gamma_3 B\lambda_j \mathbb{E}\left[\|D^{1/2}(W\Theta_t - b)\|^2 | W\right] - \gamma_1^2\gamma_3 B(B-1)\lambda_j^2\rho_j^2(t).$$

Putting this altogether yields the following system of linear ODEs.

> **Exact ODEs for $\rho_j^2, \xi_j^2, \chi_j$.** Let $(\lambda_j, \omega_j)$ be the eigenvalue-eigenvector pairs for $\check{K} = W^T DW$ and decompose $b = W\check{b} + \dot{b}$ such that $W^T D\dot{b} = 0$. Here the $(v \times v)$-matrix $D = \text{Diag}(j^{-2\alpha} : 1 \leq j \leq v)$ and $\mathscr{P}(t) \stackrel{\text{def}}{=} \mathscr{P}(\Theta_t) = \|D^{1/2}(W\Theta_t - b)\|^2$. Additionally, we drop the time in the learning rate and momentum schedules to simplify notation. The functions
> $$\rho_j^2(t) = \mathbb{E}\left[\langle \omega_j^{\otimes 2}, (\Theta_t - \check{b})^{\otimes 2}\rangle | W\right], \quad \xi_j^2(t) = \mathbb{E}\left[\langle \omega_j^{\otimes 2}, Y_t^{\otimes 2}\rangle | W\right],$$
> $$\text{and} \quad \chi_j(t) = \mathbb{E}\left[\langle \omega_j^{\otimes 2}, (\Theta_t - \check{b}) \otimes Y_t\rangle | W\right] \tag{21}$$

form a closed system of linear ODEs:

$$\frac{\mathrm{d}}{\mathrm{d}t}\nu(t;\lambda_j) = \Omega(t;\lambda_j) \times \nu(t;\lambda_j) + g(t;\lambda_j), \tag{22}$$

where $\Omega(t;\lambda_j) = \bar{\Omega}(t;\lambda_j) + \tilde{\Omega}(t;\lambda_j)$ and $g(t;\lambda_j) = \bar{g}(t;\lambda_j) + \tilde{g}(t;\lambda_j)$ such that

$$\bar{\Omega}(t;\lambda_j) \stackrel{\text{def}}{=} \begin{pmatrix} -2\gamma_2 B\lambda_j & 0 & -2\gamma_3 \\ 0 & -2\Delta & 2\gamma_1 B\lambda_j \\ \gamma_1 B\lambda_j & -\gamma_3 & -\Delta - \gamma_2 B\lambda_j \end{pmatrix},$$

$$\tilde{\Omega}(t;\lambda_j) \stackrel{\text{def}}{=} \begin{pmatrix} -2\gamma_1\gamma_3 B\lambda_j + (2\gamma_1\gamma_2\gamma_3 + \gamma_2^2 + \gamma_1^2\gamma_3^2)B(B+1)\lambda_j^2 & \gamma_3^2(1-\Delta)^2 & 2\gamma_3\Delta + 2(\gamma_2\gamma_3 + \gamma_3^2\gamma_1)(1-\Delta)B\lambda_j \\ \gamma_1^2 B(B+1)\lambda_j^2 & \Delta^2 & -2\gamma_1\Delta B\lambda_j \\ -\gamma_1\gamma_2(B(B+1))\lambda_j^2 - \gamma_1^2\gamma_3 B(B+1)\lambda_j^2 & \gamma_3\Delta(2-\Delta) & (\gamma_2\Delta - 2(1-\Delta)\gamma_1\gamma_3)B\lambda_j \end{pmatrix},$$

$$\bar{g}(t;\lambda_j) \stackrel{\text{def}}{=} \begin{pmatrix} \gamma_2^2\lambda_j B\mathbb{E}\left[\mathscr{P}(t)\,|\,W\right] \\ \gamma_1^2\lambda_j B\mathbb{E}\left[\mathscr{P}(t)\,|\,W\right] \\ 0 \end{pmatrix}, \quad \tilde{g}(t;\lambda_j) \stackrel{\text{def}}{=} \begin{pmatrix} (2\gamma_1\gamma_2\gamma_3 + \gamma_1^2\gamma_3^2)\lambda_j B\mathbb{E}\left[\mathscr{P}(t)\,|\,W\right] \\ 0 \\ (-\gamma_1\gamma_2 - \gamma_1^2\gamma_3)B\lambda_j\mathbb{E}\left[\mathscr{P}(t)\,|\,W\right] \end{pmatrix},$$

$$\text{and} \quad \nu(t;\lambda_j) \stackrel{\text{def}}{=} \begin{pmatrix} \rho_j^2(t) \\ \xi_j^2(t) \\ \chi_j(t) \end{pmatrix}.$$

(23)

The initial conditions are such that

$$\nu(0;\lambda_j) = \begin{pmatrix} \rho_j^2(0) \\ \xi_j^2(0) \\ \chi_j(0) \end{pmatrix}.$$

**Remark C.1.** *We can write the functions $(\rho_j^2, \xi_j^2, \chi_j)$ in terms of the non-zero eigenvalues and eigenvectors of the conjugate matrix $\hat{K} \stackrel{\text{def}}{=} D^{1/2}WW^T D^{1/2}$.[10] We show this equivalence below.*

**Proposition C.1** (Equivalence of $(\rho_j^2, \xi_j^2, \chi_j)$)**.** *Suppose $\hat{K} = D^{1/2}WW^T D^{1/2}$ where $D = \text{Diag}(j^{-2\alpha}, j = 1, \ldots v)$. Then the following holds*

$$\rho_j^2(t) = \mathbb{E}\left[\left.\frac{\langle u_j, (D^{1/2}(W\Theta_t - b))\rangle^2}{\lambda_j}\right| W\right], \quad \xi_j^2(t) = \mathbb{E}\left[\left.\frac{\langle u_j, D^{1/2}WY_t\rangle^2}{\lambda_j}\right| W\right],$$

$$\text{and} \quad \chi_j(t) = \mathbb{E}\left[\left.\frac{\langle u_j^{\otimes 2}, (D^{1/2}(W\Theta_t - b)) \otimes D^{1/2}WY_t\rangle}{\lambda_j}\right| W\right],$$

*where $(\lambda_j, u_j)_{j=1}^d$ are the nonzero eigenvalues/eigenvectors of $\hat{K}$.*

*Proof.* Let $D^{1/2}W = V\Sigma U^T$ be the singular value decomposition where $V = [u_1, u_2, .., u_v] \in \mathbb{R}^{v \times v}$, $U = [\omega_1, \omega_2, \ldots, \omega_d] \in \mathbb{R}^{d \times d}$, and $\Sigma \in \mathbb{R}^{v \times d}$ rectangular diagonal matrix with non-zero singular values $\sigma_j$ where $j = 1, \ldots, d$. We prove the result for $\rho_j^2(t)$ – noting that the results for $\xi_j^2$ and $\chi_j$ follow a similar argument.

We note that $W^T D\dot{b} = 0$ implies that $W^T D^{1/2}(D^{1/2}\dot{b}) = 0$. In particular, this means that $U\Sigma^T V^T D^{1/2}\dot{b} = 0$. By hitting both sides by $\Sigma U^T$, we get that

$$\Sigma U^T U\Sigma^T V^T D^{1/2}\dot{b} = 0 \quad \Rightarrow \quad \Sigma\Sigma^T V^T D^{1/2}\dot{b} = 0.$$

Thus for all nonzero singular values $\sigma_j$ (or equivalently for all nonzero eigenvalues $\lambda_j$ of $\check{K} = W^T DW$), $u_j^T D^{1/2}\dot{b} = 0$.

Additionally, we have that

$$W^T D^{1/2}u_j = U\Sigma^T V^T u_j = U\Sigma^T e_j = \sigma_j U e_j = \sigma_j \omega_j,$$

where $e_j$ is 0 except for a 1 in the $j$th position. Note this holds for all non-zero $\sigma_j$.

---

[10]These are precisely the statistics described in (4).

We have that $b = W\check{b} + \dot{b}$ where $W^T D\dot{b} = 0$. Therefore,

$$\frac{\langle u_j, (D^{1/2}(W\Theta_t - b))\rangle^2}{\lambda_j} = \frac{\langle u_j, D^{1/2}W(\Theta_t - \check{b}) - D^{1/2}\dot{b}\rangle^2}{\lambda_j}$$

$$= \frac{\langle u_j, D^{1/2}W(\Theta_t - \check{b})\rangle^2}{\lambda_j} - \frac{\langle u_j, D^{1/2}\dot{b}\rangle^2}{\lambda_j}$$

$$= \frac{(\sigma_j\langle\omega_j, (\Theta_t - \check{b})\rangle)^2}{\lambda_j}.$$

The last equality follows since $\sigma_j^2 = \lambda_j$. Taking expectations, finishes the proof. $\square$

**Remark C.2** (Initialization). *If $\Theta_0 = Y_{0-} = 0$, then $\rho_j^2(0) = \mathbb{E}\left[\frac{\langle u_j, D^{1/2}b\rangle^2}{\lambda_j}\,\Big|\,W\right]$, $\xi_j^2(0) = \chi_j(0) = 0$ for all $j = 1, \ldots, d$.*

Since the nonzero eigenvalues of $\hat{K}$ are the same as $\check{K}$, and in light of Prop. C.1, these $\{(\rho_j^2, \xi_j^2, \chi_j)\}_{j=1}^d$ satisfy the same exact ODEs as in (22). Throughout the remaining sections, we use $\{(\rho_j^2, \xi_j^2, \chi_j)\}_{j=1}^d$ defined by the (nonzero) eigenvalues/eigenvectors of $\hat{K} = D^{1/2}WW^T D^{1/2}$.

### C.1.2 Derivation of the exact Volterra equation

In this section, we derive a Volterra equation that describes the expected loss using the exact ODEs. To do so, requires abstractly solving the inhomogeneous ODEs.

To solve (22), we employ Duhamel's principle. Let $\Phi_{\lambda_j}(t, s)$ for $t \geq s$ be the solution to the IVP

$$\frac{\mathrm{d}}{\mathrm{d}t}\Phi_{\lambda_j}(t, s) = \Omega(t; \lambda_j)\Phi_{\lambda_j}(t, s) \quad \text{such that } \Phi_{\lambda_j}(s, s) = \mathrm{Id}_3 \text{ for all } 1 \leq j \leq d. \tag{24}$$

Then by Duhamel's principle, we have for every $j$

$$\nu(t; \lambda_j) = \Phi_{\lambda_j}(t, 0)\nu(0; \lambda_j) + \int_0^t \Phi_{\lambda_j}(t, s)g(s; \lambda_j)\,\mathrm{d}s.$$

As $\rho_j^2(t) = \nu(t; \lambda_j)_1$, we have that

$$\mathbb{E}\left[\mathscr{P}(t)\,|\,W\right] = \lim_{t\to\infty}\mathbb{E}\left[\mathscr{P}(t)\,|\,W\right] + \sum_{j=1}^d \lambda_j \times \rho_j^2(t)$$

$$= \lim_{t\to\infty}\mathbb{E}\left[\mathscr{P}(t)\,|\,W\right] + \sum_{j=1}^d \lambda_j\left(\Phi_{\lambda_j}(t, 0)\nu(0; \lambda_j)\right)_1 \tag{25}$$

$$+ \int_0^t \mathbb{E}\left[\mathscr{P}(s)\,|\,W\right]\left[B\sum_j \lambda_j^2 \times \Phi_{\lambda_j}(t, s)h(s)\right]_1 \mathrm{d}s.$$

Here the vector $h(s) \in \mathbb{R}^3$ is given by

$$h(s) \stackrel{\text{def}}{=} \begin{pmatrix} \gamma_2^2 + (2\gamma_1\gamma_2\gamma_3 + \gamma_1^2\gamma_3^2) \\ \gamma_1^2 \\ (-\gamma_1\gamma_2 - \gamma_1^2\gamma_3) \end{pmatrix},$$

where the learning rate schedules $\gamma_i$, $i = 1, 2, 3$, are time-dependent. We denote the components of $h(s)$ as $h_i(s)$ where $i = 1, 2, 3$. Therefore, the expected loss satisfies a Volterra equation (not necessarily convolution-type)

$$\mathbb{E}\left[\mathscr{P}(t)\,|W\right] = \mathscr{F}(t) + \int_0^t \mathscr{K}_t(s) \times \mathbb{E}\left[\mathscr{P}(s)\,|\,W\right]\mathrm{d}s, \tag{26}$$

$$\text{where} \quad \mathscr{F}(t) \stackrel{\text{def}}{=} \sum_{j=1}^d \lambda_j \times \left(\Phi_{\lambda_j}(t, 0)\right)_{11} \times \rho_j^2(0) + \lim_{t\to\infty}\mathbb{E}[\mathscr{P}(t)\,|\,W]$$

$$\text{and} \quad \mathscr{K}_s(t) \stackrel{\text{def}}{=} B\sum_j \lambda_j^2 \times \left[h_1(s)\left(\Phi_{\lambda_j}(t, s)\right)_{11} + h_2(s)\left(\Phi_{\lambda_j}(t, s)\right)_{12} + h_3(s)\left(\Phi_{\lambda_j}(t, s)\right)_{13}\right].$$

Let $\hat{K} = V\Lambda V^T$. Define $D_{\Phi_{\lambda,11}}(t,s) \stackrel{\text{def}}{=} \text{Diag}((\Phi_{\lambda_j}(t,s))_{11} \; : \; 1 \leq j \leq v)$ and $\Phi^{11}_{\hat{K}}(t,s) \stackrel{\text{def}}{=} VD_{\Phi_{\lambda,11}}(t,s)V^T$. Here we define for the $v-d$ zero eigenvalues of $\hat{K}$ to have $(\Phi_{\lambda_j}(t,s))_{11} = 1$. Where might this come from? By allowing some leniency in the definition of the ODE (24), we can view $\lambda$ as a parameter (in the case we looked at above, we set $\lambda$ to be an eigenvalue of $\check{K}$). If one plugs $\lambda = 0$ into (24), then due to the initial conditions, we get that $\Phi_{11}(t,s) \equiv 1$ for $\lambda = 0$. Using the $\rho_j^2(t)$ as defined in Prop. C.1,

$$\mathscr{F}(t) = \lim_{t \to \infty} \mathbb{E}\left[\mathscr{P}(t) \,|\, W\right] + \sum_{j=1}^{d} \lambda_j (\Phi_{\lambda_j}(t,0))_{11} \rho_j^2(0) = \langle \Phi^{11}_{\hat{K}}(t,0), (D^{1/2}(W\Theta_0 - b))^{\otimes 2}\rangle.$$

Similarly defining $D_{\Phi_{\lambda,1k}}(t,s) \stackrel{\text{def}}{=} \text{Diag}((\Phi_{\lambda_j}(t,s))_{1k} \; : \; 1 \leq j \leq v)$ and $(\Phi_{\hat{K}}(t,s))_{1k} \stackrel{\text{def}}{=} VD_{\Phi_{\lambda,1k}}(t,s)V^T$ for $k = 2,3$,

$$\begin{aligned}
\mathscr{K}_s(t) &= B \sum_j \lambda_j^2 \times \left[h_1(s)\left(\Phi_{\lambda_j}(t,s)\right)_{11} + h_2(s)\left(\Phi_{\lambda_j}(t,s)\right)_{12} + h_3(s)\left(\Phi_{\lambda_j}(t,s)\right)_{13}\right] \\
&= B \times \text{Tr}\left(\hat{K}^2\left[h_1(s)\Phi^{11}_{\hat{K}}(t,s) + h_2(s)\Phi^{12}_{\hat{K}}(t,s) + h_3(s)\Phi^{13}_{\hat{K}}(t,s)\right]\right).
\end{aligned} \tag{27}$$

Therefore we can write Volterra equation for $\hat{K}$.

From now on, we consider a specific initialization setting.

**Assumption 2** (Initialization). *We assume at initialization, $\theta_0 = \Theta_0 = 0$ and $y_{-1} = Y_{0-} = 0$.*

We thus can represent the expected loss function as a Volterra equation

$$\mathbb{E}\left[\mathscr{P}(\Theta_t) \,|\, W\right] = \underbrace{\overbrace{\mathscr{F}(t)}^{\text{forcing func.}}}_{\substack{\text{deterministic alg.}}} + \underbrace{\int_0^t \mathscr{K}_t(s) \times \mathbb{E}\left[\mathscr{P}(\Theta_s) \,|\, W\right] \, \mathrm{d}s}_{\text{stochastic noise}}. \tag{28}$$

The forcing function $\mathscr{F}(t)$ and kernel function $\mathscr{K}_t(s)$ are explicit functions of the eigenvalues of the matrix $\hat{K} = D^{1/2}WW^T D^{1/2}$ where $D = \text{Diag}(j^{-2\alpha} \; : \; 1 \leq j \leq v)$. In particular, we have that

$$\mathscr{F}(t) = \langle \Phi^{11}_{\hat{K}}(t,0), (D^{1/2}b)^{\otimes 2}\rangle$$

$$\text{and} \quad \mathscr{K}_s(t) \stackrel{\text{def}}{=} B \times \text{Tr}\left(\hat{K}^2\left[h_1(s)\Phi^{11}_{\hat{K}}(t,s) + h_2(s)\Phi^{12}_{\hat{K}}(t,s) + h_3(s)\Phi^{13}_{\hat{K}}(t,s)\right]\right)$$

with $\frac{\mathrm{d}}{\mathrm{d}t}\Phi_{\lambda_j}(t,s) = \Omega(t;\lambda_j)\Phi_{\lambda_j}(t,s)$ such that $\Phi_{\lambda_j}(s,s) = \text{Id}_3$ for all $1 \leq j \leq v$ and $\Phi^{1k}_{\hat{K}}(t,s) = V\text{Diag}((\Phi_{\lambda_j}(t,s))_{1k} \; : \; 1 \leq j \leq v)V^T$ for $k = 1,2,3$.

> **(Exact ODEs) Volterra equation for $\hat{K} = D^{1/2}WW^T D^{1/2} = V^T\Lambda V$.** Set the $(v \times v)$-matrix $D = \text{Diag}(j^{-2\alpha} \; : \; 1 \leq j \leq v)$ and $\mathscr{P}(t) \stackrel{\text{def}}{=} \mathscr{P}(\Theta_t) = \|D^{1/2}(W\Theta_t - b)\|^2$. Let $\Phi_{\lambda_j}(t,s)$ for $t \geq s$ be the solution to the IVP
>
> $$\frac{\mathrm{d}}{\mathrm{d}t}\Phi_{\lambda_j}(t,s) = \Omega(t;\lambda_j)\Phi_{\lambda_j}(t,s) \quad \text{such that } \Phi_{\lambda_j}(s,s) = \text{Id}_3 \text{ for all } 1 \leq j \leq d. \tag{29}$$
>
> Then by Duhamel's principle, we have for every $j$
>
> $$\nu(t;\lambda_j) = \Phi_{\lambda_j}(t,0)\nu(0;\lambda_j) + \int_0^t \Phi_{\lambda_j}(t,s)g(s;\lambda_j) \, \mathrm{d}s.$$

The expected loss under the iterates (Gen-Mom-SGD) satisfies a Volterra equation

$$\mathbb{E}\left[\mathscr{P}(t)\,|\,W\right] = \mathscr{F}(t) + \int_0^t \mathscr{K}_t(s) \times \mathbb{E}\left[\mathscr{P}(s)\,|\,W\right]\,\mathrm{d}s,$$

$$\text{where} \quad \mathscr{F}(t) \overset{\mathrm{def}}{=} \langle \Phi_{\hat{K}}^{11}(t,0), (D^{1/2}(W\Theta_0 - b))^{\otimes 2} \rangle,$$

$$\mathscr{K}_s(t) \overset{\mathrm{def}}{=} B \times \mathrm{Tr}\big(\hat{K}^2\big[h_1(s)\Phi_{\hat{K}}^{11}(t,s) + h_2(s)\Phi_{\hat{K}}^{12}(t,s) + h_3(s)\Phi_{\hat{K}}^{13}(t,s)\big]\big), \tag{30}$$

$$\text{and} \quad h(s) = \begin{pmatrix} \gamma_2^2 + (2\gamma_1\gamma_2\gamma_3 + \gamma_1^2\gamma_3^2) \\ \gamma_1^2 \\ (-\gamma_1\gamma_2 - \gamma_1^2\gamma_3) \end{pmatrix}.$$

Here $\Phi_{\hat{K}}^{1k}(t,s) = V\,\mathrm{Diag}((\Phi_{\lambda_j}(t,s))_{1k} \;:\; 1 \leq j \leq v)V^T$ for $k = 1, 2, 3$ and $h_i$ are the components of the vector $h$.

While these representations are easy to see from the derivation of the Volterra equation, a more useful representation of the forcing function and kernel function is through contour integrals over the spectrum $\hat{K}$. Let $\Gamma$ be a contour containing the spectrum of $\hat{K}$ (recall, $\hat{K}$ is normalized so that the largest eigenvalue of $\hat{K}$ is 1; thus $\Gamma$ is a contour containing $[0, 1]$). Then the forcing function takes the form

$$\begin{pmatrix} \text{forcing function} \\ \text{for algorithm} \end{pmatrix} \quad \mathscr{F}(t) \overset{\mathrm{def}}{=} \frac{-1}{2\pi i} \oint_\Gamma \langle (\hat{K} - z)^{-1}, (D^{1/2}b)^{\otimes 2} \rangle (\Phi_z(t,0))_{11}\,\mathrm{d}z \tag{31}$$

and the kernel function takes the form

$$\begin{pmatrix} \text{kernel function} \\ \text{for algorithm} \end{pmatrix} \quad \mathscr{K}_s(t) \overset{\mathrm{def}}{=} B \times \mathrm{Tr}\bigg(\frac{-1}{2\pi i} \oint_\Gamma z^2\big[h_1(s)(\Phi_z(t,s))_{11} + h_2(s)(\Phi_z(t,s))_{12}$$
$$+ h_3(s)(\Phi_z(t,s))_{13}\big] \times (\hat{K} - z)^{-1}\,\mathrm{d}z\bigg). \tag{32}$$

### C.1.3 Deterministic Equivalent of the Expected Loss

The forcing function $\mathscr{F}(t)$ and kernel function $\mathscr{K}_s(t)$ are random as they depend on the random matrix $W$. Moreover the expressions via contour integration show that both of these functions can be described in terms of the random matrix $\hat{K} = D^{1/2}WW^TD^{1/2}$. Indeed it is the resolvent of $\hat{K}$,

$$\mathscr{R}(\hat{K}, z) \overset{\mathrm{def}}{=} (\hat{K} - z)^{-1},$$

which plays a significant role in $\mathscr{F}$ and $\mathscr{K}$. To analyze the power law behavior of the expected loss, we remove the randomness in $\hat{K}$, i.e., the matrix $W$. We do this by finding a deterministic equivalent for the resolvent of $\hat{K}$, $\mathscr{R}(\hat{K}, z)$, using techniques from random matrix theory. Intuitively, we want to take the expectation over the random matrix $W$; though not formally true.

Formally, we define the deterministic equivalent for the resolvent $\mathscr{R}(\hat{K}, z)$, denoted by $\mathcal{R}(z)$ implicitly via a fixed point equation

$$m(z) \overset{\mathrm{def}}{=} \frac{1}{1 + \frac{1}{d}\sum_{j=1}^v \frac{j^{-2\alpha}}{j^{-2\alpha}m(z) - z}} \quad \text{where} \quad \mathcal{R}(z) \overset{\mathrm{def}}{=} \mathrm{Diag}\left(\frac{1}{j^{-2\alpha}m(z) - z} \;:\; 1 \leq j \leq v\right). \tag{33}$$

As mentioned earlier, this deterministic equivalent $\mathcal{R}(z)$ can be viewed, roughly as,

$$\mathbb{E}_W[(\hat{K} - z)^{-1}] = \mathbb{E}_W[\mathscr{R}(\hat{K}, z)] \approx \mathcal{R}(z);$$

though it is not formally the expectation over $W$.

Using this deterministic expression for the resolvent of $\hat{K}$, we define deterministic expressions for the forcing function via the contour representation of $\mathscr{F}(t)$ in (31)

$$\mathcal{F}(t) \overset{\mathrm{def}}{=} \frac{-1}{2\pi i} \oint_\Gamma \langle \mathcal{R}(z), (D^{1/2}b)^{\otimes 2} \rangle (\Phi_z(t,0))_{11}\,\mathrm{d}z \tag{34}$$

and the kernel function in (32)

$$\mathcal{K}_s(t) \stackrel{\text{def}}{=} B \times \text{Tr}\left(\frac{-1}{2\pi i} \oint_\Gamma z^2 \big[h_1(s)(\Phi_z(t,s))_{11} + h_2(s)(\Phi_z(t,s))_{12}\right.$$
$$\left. + h_3(s)(\Phi_z(t,s))_{13}\big] \times \mathcal{R}(z) \, \mathrm{d}z\right). \tag{35}$$

Additionally, the deterministic equivalent defines two measure on the real line, which come from Stieltjes inversion,

$$\mu_{\mathcal{F}}(\mathrm{d}x) \stackrel{\text{def}}{=} \lim_{\varepsilon \downarrow 0} \frac{1}{\pi} \text{Im}(\langle \mathcal{R}(x+i\varepsilon), (D^{1/2}b)^{\otimes 2}\rangle \, \mathrm{d}x$$

$$\text{and} \quad \mu_{\mathcal{K}}(\mathrm{d}x) \stackrel{\text{def}}{=} \lim_{\varepsilon \downarrow 0} \frac{1}{\pi} \text{Im}\left(\text{Tr}(\mathcal{R}(x+i\varepsilon)x^2\right) \, \mathrm{d}x. \tag{36}$$

Using these measures, we can define the forcing and kernel functions on the real line with the forcing function defined as

$$\left(\begin{array}{c}\text{forcing function}\\\text{deterministic equivalent}\end{array}\right) \quad \mathcal{F}(t) = \int_{\mathbb{R}} (\Phi_x(t,0))_{11} \, \mu_{\mathcal{F}}(\mathrm{d}x) \tag{37}$$

and the kernel function defined as

$$\left(\begin{array}{c}\text{kernel function}\\\text{deterministic}\\\text{equivalent}\end{array}\right) \mathcal{K}_s(t) = B \times \int_{\mathbb{R}} \big[h_1(s)(\Phi_x(t,s))_{11} + h_2(s)(\Phi_x(t,s))_{12}$$
$$+ h_3(s)(\Phi_x(t,s))_{13}\big]\mu_{\mathcal{K}}(\mathrm{d}x). \tag{38}$$

Using the deterministic expressions for the forcing function $\mathcal{F}$ and kernel function $\mathcal{K}$, we define the deterministic function $\mathcal{P} : \mathbb{R} \to \mathbb{R}$ as the solution to the Volterra equation:

$$\mathcal{P}(t) = \mathcal{F}(t) + \int_0^t \mathcal{K}_s(t) \times \mathcal{P}(s) \, \mathrm{d}s. \tag{39}$$

Moreover, we know quite a bit about these two measure $\mu_{\mathcal{K}}$ and $\mu_{\mathcal{F}}$ from [80]. See Section D for details. In Section C.3, we provide some background information on solving Volterra equations.

## C.2 Simplified ODEs and simplified Volterra

In general, it is quite difficult to analyze the exact ODEs in (22). Instead, as way to analyze the scaling laws of (Gen-Mom-SGD), we derive a system of ODEs using an SDE. These SDEs were studied in [78] on a high-dimensional least squares problem and were shown numerically to reproduce the learning dynamics of the stochastic algorithms (Gen-Mom-SGD) for a variety of learning rate and momentum schedules. We will use these SDEs (and more importantly, the ODEs, denoted by *simplified ODEs*), to analyze SGD-M, DANA-constant, and DANA-decay in Section G, Section H, Section I, respectively.

Letting $\check{K} = W^T D W$, we consider the following

$$\mathrm{d}Y_t = -\delta(t)Y_t + \gamma_1(t)\big(B\nabla\mathscr{P}(\Theta_t) + \sqrt{B\mathscr{P}(\Theta_t)\check{K}} \, \mathrm{d}\mathcal{B}_t^{(1)}\big)$$
$$\mathrm{d}\Theta_t = -\gamma_3(t)Y_t - \gamma_2(t)\big(B\nabla\mathscr{P}(\Theta_t) + \sqrt{B\mathscr{P}(\Theta_t)\check{K}} \, \mathrm{d}\mathcal{B}_t^{(2)}\big), \tag{40}$$

where the initial conditions given by $\Theta_0 = \theta_0, Y_0 = y_0$, and $(\mathcal{B}_t^{(1)}, \mathcal{B}_t^{(2)} : t \geq 0)$ are two independent $d$-dimensional standard Brownian motions. We note here that $\nabla\mathscr{P}(\Theta_t) = W^T D(W\Theta_t - b)$.

### C.2.1 Derivation of the simplified ODEs.

In this section, we derive a system of ODEs that describe the behavior of the expected loss under the SDEs (40). We denote this system of ODEs as *simplified ODEs*. This system will be used to analyze SGD-M, Sec. G, DANA-constant, Sec. H and DANA-decaying, Sec I. To this end, we

suppose $\{(\lambda_j, \omega_j)\}_{j=1}^d$ are the eigenvalue/eigenvector pairs of $\check{K}$ and we write $b = W\check{b} + \dot{b}$ where $W^T D \cdot b = 0$.

As before, we will consider the follow special statistics/functions for each eigenvector of $\check{K}$

$$\rho_j^2(t) = \mathbb{E}\left[\langle \omega_j^{\otimes 2}, (\Theta_t - \check{b})^{\otimes 2}\rangle | W\right], \qquad \xi_j^2(t) = \mathbb{E}\left[\langle \omega_j^{\otimes 2}, Y_t^{\otimes 2}\rangle | W\right],$$
$$\text{and} \quad \chi_j(t) = \mathbb{E}\left[\langle \omega_j^{\otimes 2}, (\Theta_t - \check{b}) \otimes Y_t\rangle | W\right]. \tag{41}$$

Instead of using Wick's formula, we now apply Itô Calculus. First, consider the functions $\Pi_j \overset{\text{def}}{=} \langle \omega_j, \Theta_t - \check{b}\rangle$, and $\Xi_j \overset{\text{def}}{=} \langle \omega_j, Y_t\rangle$. A simple computation yields for $j = 1, \ldots, d$,

$$\mathrm{d}\langle \omega_j, Y_t\rangle = \langle \omega_j, \mathrm{d}Y_t\rangle = (-\delta \Xi_j + \gamma_1 B \lambda_j \Pi_j)\, \mathrm{d}t + \gamma_1 \sqrt{B\mathscr{P}(\Theta_t)}\sqrt{\lambda_j}\, \mathrm{d}\mathcal{B}_{t,j}^{(1)}$$
$$\mathrm{d}\langle \omega_j, \Theta_t - \check{b}\rangle = \langle \omega_j, \mathrm{d}\Theta_t\rangle = (-\gamma_3 \Xi_j - \gamma_2 B \lambda_j \Pi_j)\, \mathrm{d}t - \gamma_2 \sqrt{B\mathscr{P}(\Theta_t)}\sqrt{\lambda_j}\, \mathrm{d}\mathcal{B}_{t,j}^{(2)}.$$

Here $(\mathcal{B}_{t,j}^{(1)}, \mathcal{B}_{t,j}^{(2)} : t \geq 0)$ are two 1-dimensional Brownian motions. Applying Itô Calculus,

$$\mathrm{d}(\langle \omega_j, Y_t\rangle)^2 = \mathrm{d}\Xi_j^2 = 2(\Xi_j \times \left((-\delta \Xi_j + \gamma_1 B \lambda_j \Pi_j)\, \mathrm{d}t + \gamma_1 \sqrt{B\mathscr{P}(\Theta_t)}\sqrt{\lambda_j}\, \mathrm{d}\mathcal{B}_{t,j}^{(1)}\right)$$
$$+ 2\gamma_1^2 \lambda_j B\mathscr{P}(\Theta_t)\, \mathrm{d}t$$
$$\mathrm{d}(\langle \omega_j, \Theta_t - \check{b}\rangle)^2 = \mathrm{d}\Pi_j^2 = 2(\Pi_j \times \left((-\gamma_3 \Xi_j - \gamma_2 B \lambda_j \Pi_j)\, \mathrm{d}t - \gamma_2 \sqrt{B\mathscr{P}(\Theta_t)}\sqrt{\lambda_j}\, \mathrm{d}\mathcal{B}_{t,j}^{(2)}\right)$$
$$+ \gamma_2^2 \lambda_j B\mathscr{P}(\Theta_t)\, \mathrm{d}t.$$

As for the cross term, we have

$$\mathrm{d}(\langle \omega_j, Y_t\rangle\langle \omega_j, \Theta_t - \check{b}\rangle) = \mathrm{d}(\Xi_j \Pi_j) = -\gamma_3 \Xi_j^2 - \gamma_2 B \lambda_j \Xi_j \Pi_j - \delta \Xi_j \Pi_j + \gamma_1 B \lambda_j \Pi_j^2$$
$$- \gamma_2 \Xi_j \sqrt{B\mathscr{P}(\Theta_t)}\sqrt{\lambda_j}\, \mathrm{d}\mathcal{B}_{t,j}^{(2)}$$
$$+ \Pi_j \gamma_1 \sqrt{B\mathscr{P}(\Theta_t)}\sqrt{\lambda_j}\, \mathrm{d}\mathcal{B}_{t,j}^{(1)}.$$

Now taking expectations conditioned on $W$, we have that $\mathbb{E}[\Pi_j^2 | W] = \rho_j^2, \mathbb{E}[\Xi_j^2 | W] = \xi_j^2$ and $\mathbb{E}[\Pi_j \Xi_j | W] = \chi_j$. Thus,

$$\rho_j^2 = -2\gamma_2 B \lambda_j \rho_j^2 - 2\gamma_3 \chi_j + \gamma_2^2 \lambda_j B\mathbb{E}[\mathscr{P}(\Theta_t) | W]$$
$$\xi_j^2 = -2\delta \xi_j^2 + 2\gamma_1 B \lambda_j \chi_j + \gamma_1^2 \lambda_j B\mathbb{E}[\mathscr{P}(\Theta_t) | W]$$
$$\chi_j = -\gamma_3 \xi_j^2 - \gamma_2 B \lambda_j \chi_j - \delta \chi_j + \gamma_1 B \lambda_j \rho_j^2.$$

Putting this altogether yields the following system of linear ODEs, denoted by *Simplified ODEs*.

> **Simplified ODEs for $\rho_j^2, \xi_j^2, \chi_j$.** Let $(\lambda_j, \omega_j)$ be the eigenvalue-eigenvector pairs for $\check{K} = W^T DW$ and decompose $b = W\check{b} + \dot{b}$ such that $W^T D\dot{b} = 0$. Here the $(v \times v)$-matrix $D = \mathrm{Diag}(j^{-2\alpha} : 1 \leq j \leq v)$ and $\mathscr{P}(t) \overset{\text{def}}{=} \mathscr{P}(\Theta_t) = \|D^{1/2}(W\Theta_t - b)\|^2$. Additionally, we drop the time in the learning rate and momentum schedules to simplify notation. The functions
>
> $$\rho_j^2(t) = \mathbb{E}\left[\langle \omega_j^{\otimes 2}, (\Theta_t - \check{b})^{\otimes 2}\rangle | W\right], \quad \xi_j^2(t) = \mathbb{E}\left[\langle \omega_j^{\otimes 2}, Y_t^{\otimes 2}\rangle | W\right],$$
> $$\text{and} \quad \chi_j(t) = \mathbb{E}\left[\langle \omega_j^{\otimes 2}, (\Theta_t - \check{b}) \otimes Y_t\rangle | W\right] \tag{42}$$
>
> form a closed system of linear ODEs:
>
> $$\frac{\mathrm{d}}{\mathrm{d}t}\nu(t; \lambda_j) = \Omega(t; \lambda_j) \times \nu(t; \lambda_j) + g(t; \lambda_j), \tag{43}$$

where
$$\Omega(t;\lambda_j) \stackrel{\text{def}}{=} \begin{pmatrix} -2\gamma_2 B\lambda_j & 0 & -2\gamma_3 \\ 0 & -2\Delta & 2\gamma_1 B\lambda_j \\ \gamma_1 B\lambda_j & -\gamma_3 & -\Delta - \gamma_2 B\lambda_j \end{pmatrix},$$

$$g(t;\lambda_j) \stackrel{\text{def}}{=} \begin{pmatrix} \gamma_2^2 \lambda_j B\mathbb{E}\left[\mathscr{P}(t)\,|\,W\right] \\ \gamma_1^2 \lambda_j B\mathbb{E}\left[\mathscr{P}(t)\,|\,W\right] \\ 0 \end{pmatrix} \quad \text{and} \quad \nu(t;\lambda_j) \stackrel{\text{def}}{=} \begin{pmatrix} \rho_j^2(t) \\ \xi_j^2(t) \\ \chi_j(t) \end{pmatrix}.$$

(44)

Under Assumption 2, the initial conditions are such that
$$\nu(0;\lambda_j) = \begin{pmatrix} \rho_j^2(0) \\ \xi_j^2(0) \\ \chi_j(0) \end{pmatrix} = \begin{pmatrix} \mathbb{E}\left[\langle \omega_j^2, \check{b}^{\otimes 2}\rangle\,|\,W\right] \\ 0 \\ 0 \end{pmatrix}.$$

### C.2.2 Simplified ODE as the large time limit of a coin-flip algorithm

In the previous paragraph Section C.2.1 we have seen that the simplified ODE (43) models the risk under a particular system of ODEs (40). Although precise, this characterization has obvious limitations. Indeed, the SDE formulation requires the learning rates to vanish, with a correct scaling with the time and dimension. This hence does not model a practical algorithm which makes any discussion about compute in the scaling laws.

Instead, in the following we show that we can also view the simplified ODEs in (43) as dropping terms that arise due to the higher-order moments of the Gaussian data $x$ in the ODEs of a coin-flip algorithm.

The *coin-flip algorithm* is a simple two-staged version of (13) where at each iteration we flip a coin and update either the momentum iterate $Y$ or the gradient update $\Theta$. Let $N_t^y$, $N_t^\theta$ be iid Poisson processes (i.e, the coin flips) and let the initialization be such that $\Theta_0 = \theta_0$ and $Y_{0-} = y_{-1}$. At a jump of $N_t^y, N_t^\theta$, we generate $(x^i, \tilde{x}^i)_{i=1}^B$ new data points whose coordinates follow a power law with parameter $\alpha$ and update by

$$Y_t = (1 - \Delta(t))Y_{t-} + \gamma_1(t) \times \sum_{i=1}^B W^T x_{N_t}^i (x_{N_t}^i)^T (W\Theta_{t-} - b) \stackrel{\text{def}}{=} Y_{t-} + \Delta_t^y(Y_{t-}, \Theta_{t-})$$

$$\Theta_t = \Theta_{t-} - \gamma_2(t) \times \sum_{i=1}^B W^T x_{N_t}^i (x_{N_t}^i)^T (W\Theta_{t-} - b) - \gamma_3(t) \times Y_t \stackrel{\text{def}}{=} \Theta_{t-} + \Delta_t^\theta(Y_t, \Theta_{t-}).$$

(45)

We naturally extended time $t$ to be a continuous parameter with $Y_{t-} = Y_{t-1}$. This is a standard way to embed discrete processes into a continuous process.

For any $(\Theta, Y) \mapsto f(\Theta, Y) \in \mathbb{R}$, the mean behavior is given by

$$\frac{\mathrm{d}}{\mathrm{d}t}\mathbb{E}\left[f(Y_t, \Theta_t)\right] = \mathbb{E}\left[f(Y_t + \Delta_t^y(Y_t, \Theta_t), \Theta_t + \Delta_t^\theta(Y_t, \Theta_t)) - f(Y_t, \Theta_t)\right].$$

(46)

**Derivation of the coin-flip ODEs.** In this paragraph, we derive the system of ODEs that describe the behavior of the expected loss under the coin-flip algorithm. For this, we need to introduce statistics that are closed under differentiation as defined in (46).

As previously, we begin by writing $\mathbb{R}^v = \text{Im}(W) \oplus W^\perp$. Thus, there exists a $\check{b} \in \mathbb{R}^d$ and $\dot{b} \in \mathbb{R}^v$ such that one can write $b = W\check{b} + \dot{b}$, that is, we can decompose $b$ as an element in the image of $W$ and an element in the co-ker of $W$. Formally, we have that

$$b = W\check{b} + \dot{b}, \quad \text{where } W^T D\dot{b} = 0.$$

Indeed, denote $\bar{b} \in \text{Im}(W)^\perp$ with $b - \bar{b} \in \text{Im}(W)$. Then we are looking for some $\dot{b} \in \text{Im}(W) + \{\bar{b}\}$ satisfying $W^T D\dot{b} = 0 \iff D\dot{b} \in \text{Im}(W)^\perp \iff D\bar{b} \in \text{Im}(W)^\perp + D\,\text{Im}(W)$. Hence such $\dot{b}$ exists in particular if $D\,\text{Im}(W) \oplus \text{Im}(W)^\perp = \mathbb{R}^v$. This is clearly the case since $\dim(D\,\text{Im}(W)) = \dim(\text{Im}(W)) = v - \dim(\text{Im}(W)^\perp)$ and because $D$ never sends a vector $x$ on a perpendicular one, ie $\forall x \in \mathbb{R}^v$, $\langle x, Dx\rangle = 0 \iff x = 0$ since it is diagonal with strictly positive eigenvalues.

Letting $\check{K} \stackrel{\text{def}}{=} W^T D W \in \mathbb{R}^{d \times d}$ and $(\lambda_j, \omega_j)_{j=1}^d$ by the eigenvalue-eigenvector pairs for $\check{K}$, we consider the same special functions as in (20),

$$\rho_j^2(t) = \mathbb{E}\left[\langle \omega_j^{\otimes 2}, (\Theta_t - \check{b})^{\otimes 2}\rangle | W\right], \qquad \xi_j^2(t) = \mathbb{E}\left[\langle \omega_j^{\otimes 2}, Y_t^{\otimes 2}\rangle | W\right],$$
$$\text{and} \quad \chi_j(t) = \mathbb{E}\left[\langle \omega_j^{\otimes 2}, (\Theta_t - \check{b}) \otimes Y_t\rangle | W\right].$$
(47)

As before, knowing the $\rho_j^2$'s suffices to recover the expected loss under the coin-flip algorithm. We proceed like in Section C.1 using Wick's rule to get a closed formula for $(\rho_j^2, \xi_j^2, \chi_j)$.

$\underline{\rho_j^2 \text{ functions:}}$  Using (46) applied to $\mathbb{E}\left[f(\Theta_t, Y_t)\right] = \rho_j^2(t)$ yields,

$$\frac{\mathrm{d}}{\mathrm{d}t}\rho_j^2(t) = 2\mathbb{E}\left[\langle \omega_j^{\otimes 2}, \left(-\gamma_2(t) \times \sum_{i=1}^B W^T \tilde{x}_t^i(\tilde{x}_t^i)^T(W\Theta_t - b) - \gamma_3(t)Y_t\right) \otimes (\Theta_t - \check{b})\rangle | W\right]$$

$$+ \mathbb{E}\left[\langle \omega_j^{\otimes 2}, \left(\gamma_2(t) \times \sum_{i=1}^B W^T \tilde{x}_t^i(\tilde{x}_t^i)^T(W\Theta_t - b)\right)^{\otimes 2}\rangle | W\right] + \gamma_3^2(t)\mathbb{E}\left[\langle \omega_j^{\otimes 2}, Y_t^{\otimes 2}\rangle | W\right]$$

$$+ 2\gamma_3(t)\mathbb{E}\left[\langle \omega_j^{\otimes 2}, \left(-\gamma_2 \sum_{i=1}^B W^T \tilde{x}_{t,i}\tilde{x}_{t,i}^T(W\Theta_t - b)\right) \otimes (-Y_t)\rangle | W\right]$$

$$= -2\gamma_2(t)B\lambda_j\rho_j^2(t) - 2\gamma_3(t)\chi_j(t) + 2\gamma_2^2(t)B\lambda_j^2\rho_j^2(t)$$
$$+ \gamma_2^2(t)\lambda_j B\mathbb{E}\left[\|D^{1/2}(W\Theta_t - b)\|^2 | W\right] + B(B-1)\gamma_2^2(t)\lambda_j^2\rho_j^2(t) + \gamma_3^2(t)\xi_j^2(t)$$
$$+ 2\gamma_3(t)\gamma_2(t)B\lambda_j\chi_j(t).$$

$\underline{\xi_j^2 \text{ functions:}}$ Using (46) applied to $\mathbb{E}\left[f(\Theta_t, Y_t)\right] = \xi_j^2(t)$ yields,

$$\frac{\mathrm{d}}{\mathrm{d}t}\xi_j^2(t) = \mathbb{E}\left[\langle \omega_j^{\otimes 2}, \left((1 - \Delta(t))Y_t + \gamma_1(t)\sum_{i=1}^B W^T x_t^i(x_t^i)^T(W\Theta_t - b)\right)^{\otimes 2}\rangle | W\right]$$

$$- \mathbb{E}\left[\langle \omega_j^{\otimes 2}, Y_t^{\otimes 2}\rangle | W\right]$$
$$= 2\mathbb{E}\left[\langle \omega_j^{\otimes 2}, Y_t \otimes -\Delta(t)Y_t\rangle | W\right] + 2B\mathbb{E}\left[\langle \omega_j^{\otimes 2}, Y_t \otimes \gamma_1(t)\left(W^T x_t x_t^T(W\Theta_t - b)\right)\rangle | W\right]$$
$$+ \mathbb{E}\left[\langle \omega_j^{\otimes 2}, \left(\Delta(t)Y_t\right)^{\otimes 2}\rangle | W\right]$$
$$- 2B\mathbb{E}\left[\langle \omega_j^{\otimes 2}, \Delta(t)Y_t \otimes \gamma_1(t)\left(W^T x_t x_t^T(W\Theta_t - b)\right)\rangle | W\right]$$
$$+ \gamma_1^2(t)B\mathbb{E}\left[\langle \omega_j^{\otimes 2}, \left(W^T x_t x_t^T(W\Theta_t - b)\right)^{\otimes 2}\rangle | W\right]$$
$$+ \gamma_1^2(t)B(B-1)\mathbb{E}\left[\langle \omega_j^{\otimes 2}, \left(W^T x_t^1(x_t^1)^T(W\Theta_t - b)\right) \otimes \left(W^T x_t^2(x_t^2)^T(W\Theta_t - b)\right)\rangle | W\right]$$
$$= -2\Delta(t)\xi_j^2(t) + 2\gamma_1(t)B\lambda_j\chi_j(t) + \Delta^2(t)\xi_j^2(t) - 2\Delta(t)\gamma_1(t)B\lambda_j\chi_j(t)$$
$$+ 2\gamma_1^2(t)B\lambda_j^2\rho_j^2(t) + \gamma_1^2(t)B\lambda_j\mathbb{E}\left[\|D^{1/2}(W\Theta_t - b)\|^2 | W\right] + B(B-1)\gamma_1^2(t)\lambda_j^2\rho_j^2(t).$$

$\underline{\chi_j \text{ functions:}}$ Lastly, using (19) applied to $\mathbb{E}\left[f(\Theta_t, Y_t)\right] = \chi_j$ yields,

$$\frac{\mathrm{d}}{\mathrm{d}t}\mathbb{E}\left[\chi_j \mid W\right] = \mathbb{E}\left[\langle \omega_j^{\otimes 2}, (\Theta_t - \check{b}) \otimes \left((1-\Delta(t))Y_t + \gamma_1(t)\sum_{i=1}^{B} W^T x_t^i (x_t^i)^T (W\Theta_t - b)\right)\rangle \mid W\right]$$

$$- \mathbb{E}\left[\langle \omega_j^{\otimes 2}, (\Theta_t - \check{b}) \otimes Y_t \rangle \mid W\right]$$

$$+ \mathbb{E}\left[\langle \omega_j^{\otimes 2}, \Theta_t - \gamma_2 \sum_{i=1}^{B} W^T \tilde{x}_t^i (\tilde{x}_t^i)^T (W\Theta_t - b) - \gamma_3(t)Y_t - \check{b} \otimes Y_t \rangle \mid W\right]$$

$$- \mathbb{E}\left[\langle \omega_j^{\otimes 2}, (\Theta_t - \check{b}) \otimes Y_t \rangle \mid W\right]$$

$$= -\Delta(t)\mathbb{E}\left[\langle \omega_j^{\otimes 2}, (\Theta_t - \check{b}) \otimes Y_t \rangle \mid W\right]$$

$$+ B\mathbb{E}\left[\langle \omega_j^{\otimes 2}, (\Theta_t - \check{b}) \otimes \gamma_1(t)\left(W^T x_t x_t^T (W\Theta_t - b)\right)\rangle \mid W\right]$$

$$- \gamma_2(t)B\mathbb{E}\left[\langle \omega_j^{\otimes 2}, \left(W^T \tilde{x}\tilde{x}^T(W\Theta_t - b)\right), Y_t \rangle \mid W\right] - \gamma_3(t)\mathbb{E}\left[\langle \omega_j^{\otimes 2}, Y_t^{\otimes 2}\rangle \mid W\right]$$

$$= -\Delta(t)\chi_j(t) + \gamma_1(t)B\lambda_j \rho_j^2(t) - \gamma_2(t)B\lambda_j \chi_j(t) - \gamma_3(t)\xi_j^2(t).$$

Putting this altogether yields the following system of linear ODEs.

> **Coin-flip ODEs for $\rho_j^2, \xi_j^2, \chi_j$.** Let $(\lambda_j, \omega_j)$ be the eigenvalue-eigenvector pairs for $\check{K} = W^T DW$ and decompose $b = W\check{b} + \check{b}$ such that $W^T D\dot{b} = 0$. Here the $(v \times v)$-matrix $D = \mathrm{Diag}(j^{-2\alpha} : 1 \le j \le v)$ and $\mathscr{P}(t) \stackrel{\mathrm{def}}{=} \mathscr{P}(\Theta_t) = \|D^{1/2}(W\Theta_t - b)\|^2$.
>
> $$\rho_j^2(t) = \mathbb{E}\left[\langle \omega_j^{\otimes 2}, (\Theta_t - \check{b})^{\otimes 2}\rangle | W\right], \quad \xi_j^2(t) = \mathbb{E}\left[\langle \omega_j^{\otimes 2}, Y_t^{\otimes 2}\rangle | W\right],$$
> $$\text{and} \quad \chi_j(t) = \mathbb{E}\left[\langle \omega_j^{\otimes 2}, (\Theta_t - \check{b}) \otimes Y_t \rangle | W\right] \tag{48}$$
>
> form a closed system of linear ODEs:
>
> $$\frac{\mathrm{d}}{\mathrm{d}t}\nu(t; \lambda_j) = \Omega(t; \lambda_j) \times \nu(t; \lambda_j) + g(t; \lambda_j), \tag{49}$$
>
> where
>
> $$\Omega(t; \lambda_j) \stackrel{\mathrm{def}}{=} \begin{pmatrix} -2\gamma_2(t)B\lambda_j + B(B+1)\gamma_2^2(t)\lambda_j^2 & \gamma_3^2(t) & 2\gamma_3(t)(-1+\gamma_2(t)B\lambda_j) \\ \gamma_1^2(t)B(B+1)\lambda_j^2 & -2\Delta(t) + \Delta^2(t) & 2\gamma_1(t)B\lambda_j(1-\Delta(t)) \\ \gamma_1(t)B\lambda_j & -\gamma_3(t) & -\Delta(t) - \gamma_2(t)B\lambda_j \end{pmatrix},$$
>
> $$g(t; \lambda_j) \stackrel{\mathrm{def}}{=} \begin{pmatrix} \gamma_2^2(t)\lambda_j B\mathbb{E}\left[\mathscr{P}(t) \mid W\right] \\ \gamma_1^2(t)\lambda_j B\mathbb{E}\left[\mathscr{P}(t) \mid W\right] \\ 0 \end{pmatrix}, \text{ and } \nu(t; \lambda_j) \stackrel{\mathrm{def}}{=} \begin{pmatrix} \rho_j^2(t) \\ \xi_j^2(t) \\ \chi_j(t) \end{pmatrix}.$$
> $$\tag{50}$$
>
> The initial conditions are such that
>
> $$\nu(0; \lambda_j) = \begin{pmatrix} \rho_j^2(0) \\ \xi_j^2(0) \\ \chi_j(0) \end{pmatrix}.$$

We see that dropping some high-orders terms in the eigenvalue $\lambda \stackrel{\mathrm{def}}{=} \sigma^2$ and learning rates/momentum schedules from the ODEs (22) yields the *simplified ODEs* in (43). This is the reason why we believe that working with (43) should not affect our main results. Indeed, when studying the asymptotics for large time $t$, the contribution to the risk of the different eigenvalues will come mainly from small eigenvalues $\sigma^2$ (and especially $\sigma^2 \lesssim \frac{1}{\gamma_2 Bt}$), as for larger $\sigma$'s, solutions to the ODE have exponential decay.

**Simplified ODE as a high-dimensional limit and re-deriving the SDE system** (40) **without Itô calculus** The goal of this section is to relate the solutions of the simplified ODE Equation (43) to limits of solutions of the ODE eq. (49). This allows to re-derive without Itô calculus, some positiveness result on the sign of the simplified ODE solutions since we know the sign of the coin-flip ODE solutions.

In the following suppose that $\gamma_1(t) \equiv 1$ and that $\gamma_2(t)$, $\gamma_3(t)$, $\Delta(t)$ are three continuous functions.

Do the change of variable $\tilde{\Phi}(t) \stackrel{def}{=} \begin{pmatrix} \Phi_1(t) \\ \frac{\gamma_3(t)}{\gamma_1 B \sigma^2} \Phi_2(t) \\ \frac{\sqrt{\gamma_3(t)}}{\sqrt{\gamma_1} B \sigma} \Phi_3(t) \end{pmatrix}$ on Equation (43) and get the new ODE

$$\frac{d\tilde{\Phi}(t)}{dt} = \begin{pmatrix} -2\gamma_2 B\sigma^2 & 0 & -2\sqrt{\gamma_3\gamma_1}B\sigma \\ 0 & -2\Delta(t) + \frac{\dot{\gamma}_3(t)}{\gamma_3(t)} & 2\sqrt{\gamma_1\gamma_3}B\sigma \\ \sqrt{\gamma_1\gamma_3}B\sigma & -\sqrt{\gamma_3\gamma_1}B\sigma & -\gamma_2 B\sigma^2 - \Delta(t) + \frac{\dot{\gamma}_3(t)}{2\gamma_3(t)} \end{pmatrix} \tilde{\Phi}(t). \qquad (51)$$

**Lemma C.1.** *Let $\gamma_1(t) \equiv 1$, let $B, \sigma > 0$, and $\gamma_2(t), \gamma_3(t), \Delta(t)$ be three continuous functions with $\gamma_2(t), \gamma_3(t) > 0$. Let $\lambda \stackrel{def}{=} \sigma^2$, $s > -1$ and $M_s \in \mathcal{M}_{3\times3}(\mathbb{R})$. Denote $\Psi : (-1, \infty) \to \mathcal{M}_{3\times3}(\mathbb{R})$ the solution of the simplified ODE (51) with initial condition for $s > -1$, $\Psi(s) = M_s$. Let $\bar{d} > 0$ and denote $\Phi(t)$ the solution of the ODE (49) where we used $\tilde{\Delta}(t) \stackrel{def}{=} \frac{\Delta(t/\bar{d})}{\bar{d}}$, $\tilde{\gamma}_i(t) = \frac{\gamma_i(t/\bar{d})}{\bar{d}}$ for $i \in [3]$. Additionally denote $\hat{\Phi}(t) \stackrel{def}{=} \begin{pmatrix} \Phi_1(\bar{d}t) \\ \frac{\gamma_3(t)}{\gamma_1(t)B\sigma^2} \Phi_2(\bar{d}t) \\ \frac{\sqrt{\gamma_3(t)}}{\sqrt{\gamma_1(t)}B\sigma} \Phi_3(\bar{d}t) \end{pmatrix}$ and write the initial condition $\hat{\Phi}(s) = M_s$. Then $\forall T > -1$,*

$$\sup_{t \in [s,T]} \left\| \Psi(t) - \hat{\Phi}(t) \right\| = \mathcal{O}(\frac{1}{\bar{d}}).$$

*Proof.* Denote $\kappa = \frac{B+1}{B} \in ]1, 2]$. We obtain

$$\frac{d\hat{\Phi}(t)}{dt} = \bar{d} \begin{pmatrix} -2\frac{\gamma_2(t)B\sigma^2}{\bar{d}} + \frac{B^2\kappa\gamma_2^2(t)\sigma^4}{\bar{d}^2} & \frac{\gamma_3(t)\gamma_1(t)B\sigma^2}{\bar{d}^2} & 2\frac{\sqrt{\gamma_3(t)\gamma_1(t)B}}{\bar{d}}(-1 + \frac{\gamma_2(t)B\sigma^2}{\bar{d}}) \\ \frac{\gamma_3(t)\gamma_1(t)}{\bar{d}^2}B\kappa\sigma^2 & -2\frac{\Delta(t)}{\bar{d}} + \frac{\Delta(t)^2}{\bar{d}^2} + \frac{\dot{\gamma}_3(t)}{\gamma_3(t)} & 2\frac{\sqrt{\gamma_3(t)\gamma_1(t)B}}{\bar{d}}\sigma(1 - \frac{\Delta(t)}{\bar{d}}) \\ \frac{\sqrt{\gamma_3(t)\gamma_1(t)B}}{\bar{d}}\sigma & -\frac{\sqrt{\gamma_3(t)\gamma_1(t)B}}{\bar{d}}\sigma & -\frac{\Delta(t)}{\bar{d}} - \frac{\gamma_2(t)B}{\bar{d}}\sigma^2\hat{\Phi}(t) + \frac{\dot{\gamma}_3(t)}{2\gamma_3(t)} \end{pmatrix} \hat{\Phi}(t)$$

$$= \left( \begin{pmatrix} -2\gamma_2(t)B\sigma^2 & 0 & -2\sqrt{\gamma_3(t)\gamma_1(t)B}\sigma \\ 0 & -2\Delta(t) + \frac{\dot{\gamma}_3(t)}{\gamma_3(t)} & 2\sqrt{\gamma_3(t)\gamma_1(t)B}\sigma \\ \sqrt{\gamma_3(t)\gamma_1(t)B}\sigma & -\sqrt{\gamma_3(t)\gamma_1(t)B}\sigma & -\Delta(t) - \gamma_2(t)B\sigma^2 + \frac{\dot{\gamma}_3(t)}{2\gamma_3(t)} \end{pmatrix} + \mathcal{O}_{B,\kappa,\sigma,t}(\frac{1}{\bar{d}}) \right) \hat{\Phi}(t)$$

which implies the result. $\square$

**Remark C.3.** *Lemma C.1 shows that the simplified ODE (51) is a "high-dimensional" limit of SGD, where the learning rates are scaled inversely proportionally to the dimension with the correct scaling, as was done in [78]. Hence one will note in the following the proximity of our results to results from [78]. However, we underline that our argument in favor of using the simplified ODE (51) is because this gets rid of higher order terms in the eigenvalue $\sigma$, whose we believe the contribution to vanish for large time $t$. Indeed, we avoid taking the learning rates to vanish for large dimension, as this would make the compute-optimal scaling laws useless. Hence it is surprising that dropping higher order terms in the eigenvalues $\sigma$ (looking at the dynamics for large $t$) is equivalent to dropping higher order terms in the learning rates $\gamma_{1,2,3}$ (looking at a non-degenerate, high-dimensional limit of the algorithm for fixed time $t$). We leave the study of the links between both for future work.*

**Corollary C.1.** *Under the same conditions as in Lemma C.1 denote $\Psi : (-1, \infty) \to \mathbb{R}$ the solution of the simplified ODE (51) with initial condition for $s > -1$, $\Psi(s) = \text{Diag}(a_1, a_2, a_3)$ with $a_i > 0$ for $i \in [2]$. Then, $\forall t > -1$, $\Psi_{ij} \geq 0$ for $i, j \in [2]$.*

*Proof.* For any $\bar{d} > 0$, consider the solution $\Phi(t)$ of the ODE (49) where $\tilde{\Delta}(t) \stackrel{def}{=} \frac{\Delta(t/\bar{d})}{\bar{d}}$, $\tilde{\gamma}_i(t) = \frac{\gamma_i(t/\bar{d})}{\bar{d}}$ for $i \in [3]$ and with initial condition $\Phi(s) = \text{Diag}(a_1, a_2\frac{\bar{d}^2\sigma^2 B}{\gamma_1\gamma_3}, a_3\frac{\bar{d}\sigma\sqrt{B}}{\sqrt{\gamma_1\gamma_3}})$. We know that $(\rho_j^2(t), \xi_j^2(t), \chi_j(t))$ follows the same ODE as $\Phi$. Hence, if the initial condition is admissible, we can deduce that the first and second coefficients, being squares, are positive. We know that $\Phi_{:1}(s), \Phi_{:2}(s)$ are admissible which directly brings that $\forall t > -1, \Phi_{11}(t), \Phi_{12}(t), \Phi_{21}(t), \Phi_{22}(t) \geq 0$. Then we can pass to the limit-inf when $\bar{d} \to \infty$ by using lemma C.1 to get the same result on $\Psi$. $\square$

### C.2.3 Derivation of the simplified Volterra equation

We follow the exact same ideas as in Section C.1.2 replacing the matrix $\Omega(t; \lambda)$ in the exact ODEs (23) with $\Omega(t; \lambda_j)$ from (44). First, we let $\Phi_{\lambda_j}(t, s)$ for $t \geq s$ be the solution to the IVP

$$\frac{\mathrm{d}}{\mathrm{d}t}\Phi_{\lambda_j}(t, s) = \Omega(t; \lambda_j)\Phi_{\lambda_j}(t, s) \quad \text{such that } \Phi_{\lambda_j}(s, s) = \mathrm{Id}_3 \text{ for all } 1 \leq j \leq d. \tag{52}$$

where $\Omega(t; \lambda_j)$ is defined in (44).

Following Section C.1.2, we represent the expected loss function under the SDE (40) as a Volterra equation following an application of Duhamel's principle, that is,

$$\nu(t; \lambda_j) = \Phi_{\lambda_j}(t, 0)\nu(0; \lambda_j) + \int_0^t \Phi_{\lambda_j}(t, s)g(s; \lambda_j) \, \mathrm{d}s. \tag{53}$$

Note that in this case the function $h(s)$ simplifies,

$$h(s) \mapsto \begin{pmatrix} \gamma_2^2 \\ \gamma_1^2 \\ 0 \end{pmatrix}.$$

> **(Simplified) Volterra equation for** $\hat{K} = D^{1/2}WW^T D^{1/2} = V^T \Lambda V$**.** Set the $(v \times v)$-matrix $D = \mathrm{Diag}(j^{-2\alpha} : 1 \leq j \leq v)$ and $\mathscr{P}(t) \stackrel{\mathrm{def}}{=} \mathscr{P}(\Theta_t) = \|D^{1/2}(W\Theta_t - b)\|^2$. Let $\Phi_{\lambda_j}(t, s)$ for $t \geq s$ be the solution to the IVP
>
> $$\frac{\mathrm{d}}{\mathrm{d}t}\Phi_{\lambda_j}(t, s) = \Omega(t; \lambda_j)\Phi_{\lambda_j}(t, s) \quad \text{such that } \Phi_{\lambda_j}(s, s) = \mathrm{Id}_3 \text{ for all } 1 \leq j \leq d, \tag{54}$$
>
> where $\Omega(t; \lambda_j)$ is defined in (44). The expected loss under the iterates of the SDE (40) satisfies a Volterra equation
>
> $$\mathbb{E}\left[\mathscr{P}(t) \,|\, W\right] = \mathscr{F}(t) + \int_0^t \mathscr{K}_t(s) \times \mathbb{E}\left[\mathscr{P}(s) \,|\, W\right] \mathrm{d}s,$$
>
> $$\text{where} \quad \mathscr{F}(t) \stackrel{\mathrm{def}}{=} \langle \Phi_{\hat{K}}^{11}(t, 0), (D^{1/2}(W\Theta_0 - b))^{\otimes 2}\rangle, \tag{55}$$
>
> $$\text{and} \quad \mathscr{K}_s(t) \stackrel{\mathrm{def}}{=} B \times \mathrm{Tr}\big(\hat{K}^2\big[\gamma_2^2(s)\Phi_{\hat{K}}^{11}(t, s) + \gamma_1^2(s)\Phi_{\hat{K}}^{12}(t, s)\big]\big).$$
>
> Here $\Phi_{\hat{K}}^{1k}(t, s) = V \mathrm{Diag}((\Phi_{\lambda_j}(t, s))_{1k} : 1 \leq j \leq v)V^T$ for $k = 1, 2$.

**Deterministic equivalent for the SDEs.** As the (simplified) forcing and kernel function only depend on $\hat{K}$, like the (exact) forcing function (31) and kernel function (32), we use the same deterministic equivalent of the resolvent in (33) and the measures, $\mu_{\mathscr{F}}$ and $\mu_{\mathscr{K}}$ in (36). Therefore define the deterministic equivalences for the forcing and kernel functions of the simplified ODEs on the real line as

$$\begin{pmatrix} \text{(simplified)} \\ \text{forcing function} \\ \text{deterministic equivalent} \end{pmatrix} \quad \mathscr{F}(t) = \int_{\mathbb{R}} (\Phi_x(t, 0))_{11} \, \mu_{\mathscr{F}}(\mathrm{d}x) \tag{56}$$

and the kernel function defined as

$$\begin{pmatrix} \text{(simplified)} \\ \text{kernel function} \\ \text{deterministic} \\ \text{equivalent} \end{pmatrix} \quad \mathscr{K}_s(t) = B \times \int_{\mathbb{R}} \big[\gamma_2^2(s)(\Phi_x(t, s))_{11} + \gamma_1^2(s)(\Phi_x(t, s))_{12}\big]\mu_{\mathscr{K}}(\mathrm{d}x). \tag{57}$$

Using the deterministic expressions for the forcing function $\mathscr{F}$ and kernel function $\mathscr{K}$, we define the deterministic function $\mathscr{P} : \mathbb{R} \to \mathbb{R}$ as the solution to the Volterra equation:

$$\mathscr{P}(t) = \mathscr{F}(t) + \int_0^t \mathscr{K}_s(t) \times \mathscr{P}(s) \, \mathrm{d}s. \tag{58}$$

We know quite a bit about these two measures $\mu_{\mathcal{K}}$ and $\mu_{\mathcal{F}}$ defined in the forcing and kernel function from [80]. This knowledge allows us to derive scaling properties for the algorithms in Section B. See Section D for details. In the next section, we provide some background information on solving (general) Volterra equations.

In the rest of the appendix, we will use this simplified system of ODEs to analyze SGD-M, DANA-constant, DANA-decaying in Section G, H, I.

**Assumption 3** ((Simplified Forcing and Kernel Functions and Simplified Volterra Equation)). *From now on, we will only work with the (simplified) forcing function and (simplified) kernel function as defined in* (56) *and* (57). *The same holds for the corresponding (simplified) Volterra equation* (55). *We will drop the reference to simplified in the forcing and kernel functions and Volterra equation from now on.*

We finish this section with a remark.

**Remark C.4** (Thm. 4.1 $\hat{\mathcal{F}}(\vartheta(t))$ vs. $\mathcal{F}(t)$ in this section.). *We note the following connection with Theorem 4.1 in the introduction, which we hope does not add too much confusion. We have*

$$(\textit{Theorem. 4.1}) \quad \hat{\mathcal{F}}(\vartheta(t)) \quad \textit{corresponds to} \quad \mathcal{F}(t) \quad (\textit{Section } C, (56) \ \& \ \textit{Appendix}).$$

*In particular, the function $\mathcal{F}(t)$ (56) derived in this section and considered in the rest of the appendix, in an asymptotic sense, includes the $\vartheta(t)$ defined in Theorem 4.1. For $\mathcal{K}(t) \stackrel{\text{def}}{=} \mathcal{K}_0(t)$, it is a slightly more complicated correspondence, here*

$$(\textit{Theorem. 4.1}) \quad \hat{\mathcal{K}}(\vartheta(t)) \quad \textit{corresponds to} \quad \tfrac{1}{\gamma^2 B}\mathcal{K}(t) \quad (\textit{Section } C, (57) \ \& \ \textit{Appendix}),$$

*where $\gamma$ is defined in Theorem 4.1. This holds true also for $\mathcal{K}_s(t) \asymp \mathcal{K}_{pp}(t, s)$ which we will define properly in the next few sections.*

### C.3 Background on Volterra equations

While Volterra equations such as (58) are quite nice and well-studied in the literature (see, e.g., [45]), we need an approximation of the solution to them in order to have a better understanding of the scaling laws. In particular, we need the (deterministic equivalent) loss function $\mathcal{P}(t)$ to be a constant multiple of the forcing function $\mathcal{F}$ and kernel function $\mathcal{K}$. We state this idea more precisely at the end of the section.

To do this, we need some background on *general Volterra equations* of the form:

$$P(t) = F(t) + (K * P)(t), \quad \text{where } (K * P)(t) = \int_0^t K(t, s)P(s) \ \mathrm{d}s, \tag{59}$$

and where $F(t)$ is a non-negative forcing function and $K(t, s)$ is non-negative and monotonically increasing in second variable and monotonically decreasing in the first variable kernel function. In general, we define $(K * K)(t, s) = \int_0^t K(t, r)K(r, s) \ \mathrm{d}r$.

Let us define $K^{*n}(t, s) \stackrel{\text{def}}{=} \underbrace{(K * K * \cdots K * K)}_{n \text{ times}}(t, s)$, the $n$-fold *convolution* of $K$ where $K^{*1} =$ $K(t, s)$. Under mild assumptions such as $\|K\| \stackrel{\text{def}}{=} \sup_{t \in [0, \infty)} \int_0^t K(t, s) \ \mathrm{d}s < 1$ [11] (we call this quantity the kernel norm) and the forcing function $F$ is bounded, there exists a unique (bounded) solution $P(t)$ to (59) and the solution is given by repeatedly *convolving* the forcing function with $K$ (see, e.g., [45]),

$$P(t) = F(t) + \sum_{j=1}^{\infty}(K^{*j} * F)(t) = F(t) + (K * F)(t) + (K * K * F)(t) + (K * K * K * F)(t) + .$$

This representation of the solution to (59) enables us to get good bounds on $P(t)$. More precisely, [45, Theorem 1] shows that if $K : \mathbb{R}_+ \times \mathbb{R}_+ \to \mathbb{R}$ is continuous, $K(t, s) = 0$ for $t > s$ and $F$ bounded, then a solution to the Volterra equation is as written above. [45, Theorem 3] shows that if additionally $\limsup_{t \geq 0} \int_0^t |K(t, s)| \ \mathrm{d}s < 1$ then the above solution is bounded.

---

[11] In the case that $K$ is of convolution type, that is $K(t, s)$ can be expressed as $K(t - s)$, this definition of $\|K\|$ simplifies. Particularly, we define $\|K\| = \int_0^{\infty} K(t) \ \mathrm{d}t$.

## C.4 Reducing the complexity in the Volterra equation

First, we state and prove a lemma attributed to Kesten's Lemma [8, Lemma IV.4.7].

**Lemma C.2** (Generalized Kesten's Lemma). *Suppose $\mathcal{K}(t, s)$ is a positive kernel . Suppose that for some constant $\mathcal{C}(K)$, for all $t \geq r \geq 0$:*

$$\int_r^t K(t, s)K(s, r)ds \leq \mathcal{C}(K)K(t, r). \tag{60}$$

*Then, for all $n \geq 0$:*

$$\sup_{t \geq r \geq 0} \left\{ \frac{K^{*(n+1)}(t, r)}{K(t, r)} \right\} \leq \mathcal{C}(K)^n.$$

*Proof.* We follow the proof in [80, Lemma C.1]. Define

$$a_n \overset{\text{def}}{=} \sup_{t \geq r \geq 0} \frac{K^{*n}(t, r)}{K(t, r)\mathcal{C}(K)^{n-1}}.$$

Then it is immediate that $a_1 = 1$. If we can show that

$$a_n \leq 1,$$

then we are done. By definition of the operator we have that:

$$\frac{K^{*(n+1)}(t, r)}{\mathcal{C}(K)^n} = \int_r^t \frac{K(t, s)\mathcal{K}(s, r)}{\mathcal{C}(K)} \times \frac{K^{*n}(t, s)}{K(t, s)(\mathcal{C}(K))^{n-1}} \ \mathrm{d}s \leq a_n \times \int_r^t \frac{K(t, s)K(s, r)}{\mathcal{C}(K)} \ \mathrm{d}s.$$

By the hypothesis in (60),

$$\text{for} t \geq r \geq 0, \quad \frac{K^{*(n+1)}(t, r)}{\mathcal{C}(K)^n} \leq a_n K(t, r).$$

Hence the result is shown. $\qquad\square$

Often we will not be able to analyze directly the forcing and kernel function, $F$ and $K$ resp., in the Volterra equation. Instead, we have access to upper and lower bounds on $F$ and $K$. Using these upper and lower bounds (together with Kesten's Lemma), we can still give a non-asymptotic bound on the solution to the Volterra equation.

**Lemma C.3** (Non-asymptotic Volterra bound). *Suppose the same conditions as Lemma C.2 hold. Suppose additionally that $\mathcal{C}(\bar{K}) < 1$ and $\underline{F}(t) \leq F(t) \leq \bar{F}(t)$, $\underline{K}(t, s) \leq K(t, s) \leq \bar{K}(t, s)$ with non-negative $\underline{\mathcal{F}}, \underline{K}$. Then,*

$$\underline{F}(t) + (\underline{K} * \underline{F})(t) \leq \mathcal{P}(t) \leq \bar{F}(t) + C \times (\bar{K} * \bar{F})(t),$$

*where $C = \dfrac{1}{1 - \mathcal{C}(\bar{K})}$.*

*Proof.* This proof parallels the proof in Lemma C.2 in [80]. We include the proof for completeness. We consider the upper and lower bound separately.

*Lower bound:* Since $\underline{K}$ and $\underline{F}$ are non-negative, $\sum_{j=1}^{\infty} (\underline{K}^{*j} * \underline{F})(t) \geq (\underline{K}^{*1} * \underline{F})(t) = (\underline{K} * \underline{F})(t)$. Recall the solution to the Volterra equation takes the form.

$$P(t) = F(t) + \sum_{j=1}^{\infty} (K^{*j} * F)(t).$$

It immediately follows from $\sum_{j=1}^{\infty} (\underline{K}^{*j} * \underline{\mathcal{F}})(t) \geq (\mathcal{K} * \underline{\mathcal{F}})(t)$ the lower bound.

*Upper bound:* By Lemma C.2, there exists a $\mathcal{C}(\mathcal{K}) < 1$ such that

$$\mathcal{K}^{*j}(t, r) \leq (\mathcal{C}(\mathcal{K}))^{j-1}\mathcal{K}(t, r),$$

Hence, we have that

$$\sum_{j=1}^{\infty}(\bar{K}^{*j} * \bar{F})(t) = \sum_{j=1}^{\infty}\int_0^t \bar{K}^{*j}(t,r)\bar{F}(r)\ \mathrm{d}r \le \sum_{j=1}^{\infty}\mathcal{C}(K)^{j-1}(\bar{K} * \bar{F})(t)$$

$$= \left(\frac{1}{1-\mathcal{C}(K)}\right)(\bar{K} * \bar{F})(t).$$

The upper bound is thus shown. □

**A main tool for analyzing the deterministic equivalent.** We are now state one of the main tools used to analyze the deterministic equivalent loss function (58),

$$\mathcal{P}(t) = \mathcal{F}(t) + \int_0^t \mathcal{K}_s(t) \times \mathcal{P}(s)\ \mathrm{d}s.$$

For each algorithm, we will show that the forcing function $\mathcal{F}$ and kernel function $\mathcal{K}_s(t)$ have upper and lower bounds, that is,

$$\underline{\mathcal{F}}(t) \le \mathcal{F}(t) \le \overline{\mathcal{F}}(t) \quad \text{and} \quad \underline{\mathcal{K}}_s(t) \le \mathcal{K}_s(t) \le \overline{\mathcal{K}}_s(t). \tag{61}$$

Moreover, we will show that the scaling laws for $\underline{\mathcal{F}}$ and $\overline{\mathcal{F}}$ (same for $\mathcal{K}_s(t)$) are the same. Therefore, for each algorithm separately (with the expection of DANA-decaying), we will show the following

**Informal Theorem.** *[Reduction of the Volterra Equation]* Let $\mathcal{K}(t) \overset{\text{def}}{=} \mathcal{K}_0(t)$ and define similarly $\overline{\mathcal{K}}(t)$ and $\underline{\mathcal{K}}(t)$. Suppose $2\alpha + 2\beta > 1$, $\alpha + 1 > \beta$, and $\alpha > \frac{1}{4}$[12]. For various assumptions specific to the individual algorithms, there exists an $M > 0$ large enough and constants $\tilde{C}(\alpha, \beta, M, \mathrm{alg}), \tilde{c}(\alpha, \beta, M, \mathrm{alg})$ such that if $\gamma B t > M$ then:

$$\tilde{c} \times \left(\underline{\mathcal{F}}(t) + \frac{1}{\gamma B}\underline{\mathcal{K}}(t)\right) \le \mathcal{P}(t) \le \tilde{C} \times \left(\overline{\mathcal{F}}(t) + \frac{1}{\gamma B}\overline{\mathcal{K}}(t)\right). \tag{62}$$

We denote $\gamma \overset{\text{def}}{=} \gamma_2 + \frac{\gamma_3}{\delta}$ for classic momentum and $\gamma \overset{\text{def}}{=} \gamma_2$ for both DANA algorithms and SGD.

We prove versions of this informal theorem, see Theorem G.1 for SGD-M, see Theorem H.3 for DANA-constant, see Proposition F.9 for SGD with decaying learning rate schedules.

For the upper/lower bound on the kernel and forcing functions (61) for each algorithm are given in Section F for SGD, Section G for SGD-M, Section H for DANA-constant, and Section I for DANA-decaying. The general asymptotics for the forcing and kernel function can be found in Section D.3 and Section D.4 with specifics for each algorithm in the individual algorithm sections.

# D    Measure of the Deterministic Equivalent

In light of the Informal Theorem above, to understand scaling laws, we need to analyze the (upper/lower bounds) forcing and kernel functions under the deterministic equivalent for $\hat{K}$. We recall the (simplified) forcing function, $\mathcal{F}(t)$ defined in (56) by

$$\mathcal{F}(t) = \int_0^{\infty}(\Phi_x(t,0))_{11}\mu_{\mathcal{F}}(\mathrm{d}x),$$

and the (simplified) kernel function, $\mathcal{K}_s(t)$ in (57)

$$\mathcal{K}_s(t) = B \times \int_0^{\infty}\big[\gamma_2^2(s)(\Phi_x(t,s))_{11} + \gamma_1^2(s)(\Phi_x(t,s))_{12}\big]\mu_{\mathcal{K}}(\mathrm{d}x),$$

where the measures $\mu_{\mathcal{F}}(\mathrm{d}x)$ and $\mu_{\mathcal{K}}(\mathrm{d}x)$ are defined in (36) via the Stieltjes inversion,

$$\mu_{\mathcal{F}}(\mathrm{d}x) = \lim_{\varepsilon\downarrow 0}\frac{1}{\pi}\mathrm{Im}(\langle\mathcal{R}(x+i\varepsilon),(D^{1/2}b)^{\otimes 2}\rangle)\ \mathrm{d}x$$

$$\text{and} \quad \mu_{\mathcal{K}}(\mathrm{d}x) = \lim_{\varepsilon\downarrow 0}\frac{1}{\pi}\mathrm{Im}(x^2\mathrm{Tr}(\mathcal{R}(x+i\varepsilon)))\ \mathrm{d}x.$$

$$\tag{63}$$

---

[12]While we need $\alpha > \frac{1}{4}$ and $\alpha + 1 > \beta$ formally, we believe they are only the result of the proof technique and not necessary. We believe the result would hold without these conditions.

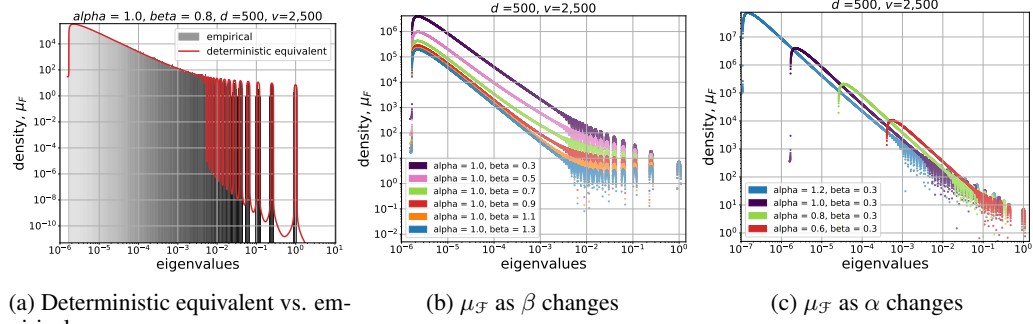

(a) Deterministic equivalent vs. empirical

(b) $\mu_{\mathcal{F}}$ as $\beta$ changes

(c) $\mu_{\mathcal{F}}$ as $\alpha$ changes

Figure 11: **Deterministic equivalent of the forcing function measure $\mu_{\mathcal{F}}$.** Numerical set-up: 100 randomly generated $\hat{K} = DWW^T D$ and the $\rho_j$'s computed; for empirical density $\mu_{\mathcal{F}}$, 500 bins equal spaced on log scale from $10^{-8}$ to 1 and counted the number of $\lambda_j$ that fall into each bin weighted by $\rho_j$'s and then averaged over the 500; For deterministic equivalent $\mu_{\mathcal{F}}$, solved fixed pointed equation (33) using Newton Method on a grid of $x$-values. Fig. 11a: deterministic density of $\mu_{\mathcal{F}}$ (63) under the deterministic equivalent for $\hat{K}$ (red) matches the empirical density of $\mu_{\mathcal{F}}$. Fig 11b: slopes of the densities $\mu_{\mathcal{F}}$ for all $\beta$ agree at the left edge, but the slopes diverge on the right edge as $\beta$ changes. Number of 'point masses' on the right edge are the same for all $\beta$. Fig. 11c: Left edge cutoff evolves like $d^{-2\alpha}$ as illustrated here. Number of 'point masses' and slope of the density change as $\alpha$ changes. These slopes effect the scaling laws.

Here $\mathcal{R}(z)$ is the deterministic equivalent of the resolvent of $\hat{K}$, given by the fixed point equation

$$m(z) = \frac{1}{1 + \frac{1}{d} \sum_{j=1}^{v} \frac{j^{-2\alpha}}{j^{-2\alpha} m(z) - z}} \quad \text{where} \quad \mathcal{R}(z) = \text{Diag}\left(\frac{1}{j^{-2\alpha} m(z) - z}\right).$$

For verification that the deterministic measures in (63) match the behavior of the random matrix $\hat{K}$, see Figure 11.

We begin with a first technical lemma stating that will allow us to show that $\mu_{\mathcal{F}}, \mu_{\mathcal{K}}$ put no mass on intervals $[1 + \epsilon, \infty)$ for $d$ large enough.

**Lemma D.1.** *Let $\alpha > 0$, $\alpha \neq \frac{1}{2}$ and $\epsilon > 0$. There exists $\bar{d} > 0$ such that $\forall d \geq \bar{d}$, admissible $v$, and any $z \in [1 + \epsilon, \infty)$, the fixed point equation $m(z) = \dfrac{1}{1 + \frac{1}{d} \sum_{j=1}^{v} \frac{j^{-2\alpha}}{j^{-2\alpha} m(z) - z}}$ has a real solution $m(z) \in [1 - \epsilon/2, 1 + \epsilon/2]$.*

*Proof.* The proof boils down to showing that for any $z \in [1 + \epsilon, \infty)$, the function $G(m; z) \overset{\text{def}}{=} \frac{1}{1 + \frac{1}{d} \sum_{j=1}^{v} \frac{j^{-2\alpha}}{j^{-2\alpha} m - z}}$ has a fixed point in $[1 - \epsilon/2, 1 + \epsilon/2]$ for $d$ large enough. Clearly for all $d > 0$, $G(m; z)$ is continuous in $m$ on $[1 - \epsilon/2, 1 + \epsilon/2]$. We write $\forall m \in [1 - \epsilon/2, 1 + \epsilon/2], \forall z \in [1 + \epsilon, +\infty)$

$$\left| \frac{1}{d} \sum_{j=1}^{v} \frac{j^{-2\alpha}}{j^{-2\alpha} m - z} \right| \leq \left| \frac{1}{d} \sum_{j=1}^{v} \frac{j^{-2\alpha}}{|1 + \epsilon/2 - z|} \right| \leq \frac{2}{\epsilon} \left| \frac{1}{d} \sum_{j=1}^{v} j^{-2\alpha} \right|$$

$$\leq \frac{2C(\alpha)}{\epsilon} d^{-1 + \max\{0, 1 - 2\alpha\}}.$$

The above can be made arbitrarily small for large $d$. Hence, by definition of $G(m; z)$, this brings the existence of $\bar{d}(\alpha, \epsilon)$ such that $\forall d \geq \bar{d}, \forall z \in [1 + \epsilon, \infty)$, $G(m; z)$ is a continuous self-map on $[1 - \epsilon/2, 1 + \epsilon/2]$. Hence it admits a fixed point in $[1 - \epsilon/2, 1 + \epsilon/2]$.

$\square$

## D.1 Estimating $\mu_{\mathcal{F}}$

In the following section, we provide upper and lower bounds on the deterministic equivalent measure of the forcing function $\mu_{\mathcal{F}}$ to compare it to reference measures $\mu_{\mathcal{F}_0}, \mu_{\mathcal{F}_{ac}}, \mu_{\mathcal{F}_{pp}}$ so that integrating it against any function $f$ can simplify as

$$\int_{\sigma \in \mathbb{R}_+} f(\sigma) \, \mathrm{d}\mu_{\mathcal{F}}(\sigma^2) \approx \int_{\sigma \in \mathbb{R}_+} f(\sigma) \, \mathrm{d}(\mu_{\mathcal{F}_0} + \mu_{\mathcal{F}_{ac}} + \mu_{\mathcal{F}_{pp}})(\sigma^2).$$

This is formalized in Proposition D.13 below.

We begin by noting for all $j \in [v]$ that we can define $\mu_{\mathcal{F}}^{(j)}(\mathrm{d}x)$ by Stieljes inversion, i.e. $\forall z \in \mathbb{H}$,

$$
\int_0^\infty \frac{\mu_{\mathcal{F}}^{(j)}(\mathrm{d}x)}{x - z} \overset{\text{def}}{=} \frac{1}{j^{-2\alpha}m(z) - z},
$$

$$
\text{or equivalently} \quad \mu_{\mathcal{F}}^{(j)}(\mathrm{d}x) \overset{\text{def}}{=} \lim_{\epsilon \downarrow 0} \frac{1}{\pi} \operatorname{Im}\left( \frac{1}{j^{-2\alpha}m(x + i\epsilon) - (x + i\epsilon)} \right) \mathrm{d}x.
$$

(64)

We therefore rewrite $\mu_{\mathcal{F}}(\mathrm{d}x) = \sum_{j=1}^v j^{-2\alpha - 2\beta} \mu_{\mathcal{F}}^{(j)}(\mathrm{d}x)$.

Now let's notice an important identity, which will be very useful to get upper and lower bounds on $\mu_{\mathcal{F}}$. That is for any $z = x + i\eta(x) \in \mathbb{H}$ we can rewrite

$$
\begin{aligned}
\frac{1}{\pi} \operatorname{Im}\left( \frac{1}{j^{-2\alpha}m(x + i\eta(x)) - (x + i\eta(x))} \right) &= \frac{1}{\pi} \operatorname{Im}\left( \int_0^\infty \frac{\mu_{\mathcal{F}}^{(j)}(\mathrm{d}s)}{s - (x + i\eta(x))} \right) \\
&= \int_0^\infty \frac{1}{\pi} \frac{\mu^{(j)}(\mathrm{d}s)\eta(x)}{(s - x)^2 + \eta(x)^2} \\
&= \left( \operatorname{Pois}_{\eta(x)} * \mu_{\mathcal{F}}^{(j)} \right)(x).
\end{aligned}
$$

Hence, estimating the left-hand term allows to evaluate the result of the convolution between $\mu_{\mathcal{F}}^{(j)}$ with Poisson kernels. We can hence use the bounds from [80] on a specific contour in the complex plane to estimate $\mu_{\mathcal{F}}$. For that, we will need a technical lemma comparing indicators on segments and specific combinations of Poisson kernels.

**Lemma D.2.** *Let* $\epsilon, c, \alpha > 0$. *Consider for* $u \in [d^{-2\alpha}, 1]$, $\eta(u) \overset{\text{def}}{=} (\log(1/\epsilon)/c) \max\{u^{1+1/(2\alpha)}, \frac{\pi}{2\alpha} \frac{u^{1-1/(2\alpha)}}{d}\}$ *and denote* $P(u) \overset{\text{def}}{=} \int_{x=4C/3}^{5C/3} \operatorname{Pois}_{\eta(x)}(u - x) \, \mathrm{d}x$. *There exists a constant* $\tilde{C}(\epsilon, c, \alpha)$ *independent of* $d, v$, *such that* $\forall C \in [d^{-2\alpha}, \frac{1}{2}]$, *we have*

*(i)*

$$
\forall s \in \mathbb{R}, \ \mathbb{1}_{[C, 2C]}(s) \leq \bar{C}\left( \int_{u=C}^{2C} \operatorname{Pois}_{\eta(u)}(s - u) \, \mathrm{d}u \right).
$$

*(ii)*

$$
\begin{cases}
if \ u \in [C, 2C] & P(u) \leq \tilde{C} \\
and \ if \ u \notin [C, 2C] & P(u) \leq \tilde{C} \min\left\{1, \frac{\eta(C)C}{d(u)^2}\right\} \ with \ d(u) \overset{\text{def}}{=} d(u, [\frac{4C}{3}, \frac{5C}{3}]).
\end{cases}
$$

*Proof.* (i) First note that $\forall u \in [C, 2C], \eta(u) \overset{\epsilon, c, \alpha}{\lesssim} C$. We can define $a \overset{\text{def}}{=} \inf_{u \in [C, 2C]} \eta(u)$, $b \overset{\text{def}}{=} \sup_{u \in [C, 2C]} \eta(u)$ and notice that there exists a constant $\bar{C}(\epsilon, c, \alpha)$ such that

$$
\forall C > 0 \ \forall s \in \mathbb{R}, \quad \mathbb{1}_{[C, 2C]}(s) \leq \bar{C}\left( \int_{u=C}^{2C} \operatorname{Pois}_b(s - u) \, \mathrm{d}u \right).
$$

From the definition of $\eta(u)$ which follows a power law, we know that $\frac{b}{a} \leq 2^{1+1/(2\alpha)}$. Hence this brings that $\forall s \in [-C, C], \frac{\sup_{\eta \in [a,b]} \operatorname{Pois}_\eta(s)}{\inf_{\eta \in [a,b]} \operatorname{Pois}_\eta(s)} \overset{\epsilon, c, M, \alpha}{\lesssim} 1$. This proves the result.

(ii) Similarly, we know that for potentially different constants $\bar{C}(\epsilon, c, \alpha), \tilde{C}(\epsilon, c, \alpha)$ we have

$$\forall C > 0 \; \forall s \in \mathbb{R}, \quad \left( \int_{u=C}^{2C} \mathrm{Pois}_b(s - u) \, du \right) \leq \bar{C}.$$

This brings using $\frac{\sup_{\eta \in [a,b]} \mathrm{Pois}_\eta(s)}{\inf_{\eta \in [a,b]} \mathrm{Pois}_\eta(s)} \overset{\epsilon, c, M, \alpha}{\lesssim} 1$ that if $u \in [C, 2C]$, $P(u) \leq \tilde{C}$. Additionally, notice that for a potentially larger $\tilde{C}$

$$\forall u \notin [C, 2C], \; \forall x \in [\frac{4C}{3}, \frac{5C}{3}], \quad \mathrm{Pois}_{\eta(x)}(u - x) \leq \tilde{C} \frac{\eta(C)}{d(u, [\frac{4C}{3}, \frac{5C}{3}])^2},$$

we obtain that still for a potentially larger $\tilde{C}$,

$$\forall u \notin [C, 2C] \quad P(u) \leq \tilde{C} \min \left\{ 1, \frac{\eta(C)C}{d(u)^2} \right\} \text{ with } d(u) \overset{\text{def}}{=} d(u, [\frac{4C}{3}, \frac{5C}{3}]).$$

$\square$

### D.1.1 Upper bound on $\mu_{\mathcal{F}}$

We begin by proving the upper bound by decomposing the real line into different ranges of 'singular values', $\sigma$'s: large $\sigma$'s, $\sigma = 0$, small $\sigma$'s, and intermediate $\sigma$'s.

A first observation is that $\mu_{\mathcal{F}}^{(j)}$ is a probability measure.

**Proposition D.1.** *Let $\alpha > 0$. For any $j \in [v]$, $\mu_{\mathcal{F}}^{(j)}$ is a positive measure and $\mu_{\mathcal{F}}^{(j)}(\mathbb{R}^+) = 1$.*

*Proof.* By definition, $\mu_{\mathcal{F}}^{(j)}(dx) \overset{\text{def}}{=} \lim_{\epsilon \downarrow 0} \frac{1}{\pi} \mathrm{Im} \left( \frac{1}{j^{-2\alpha} m(x+i\epsilon) - (x+i\epsilon)} \right) dx$. Since for all $z \in \mathbb{H}$, $m(z)$ is defined as the fixed point of (33) with negative imaginary part [80, Prop. E.1], it is clear that $\mu_{\mathcal{F}}^{(j)}$ is a positive measure. Additionally, since the support of $\mu_{\mathcal{F}}^{(j)}$ is bounded and $\frac{1}{j^{-2\alpha} m(z) - z} \underset{z=it, t \to \infty}{\sim} \frac{-1}{z}$ we see that $\mu_{\mathcal{F}}^{(j)}$ has mass 1. $\square$

This implies a first bound on the mass of $\mu_{\mathcal{F}}$ for large $\sigma$.

**Proposition D.2** (Upper bound for large $\sigma$'s). *Let $\alpha > 0$. Suppose $2\alpha + 2\beta > 1$. Then there is a constant $C(\alpha, \beta)$ such that $\forall M > 0$, we have*

$$\mu_{\mathcal{F}}([M, +\infty]) \leq C(\alpha, \beta).$$

*Additionally, for any $\epsilon > 0$, there exists some $\overline{d} > 0$ such that $\forall d \geq \overline{d}$, $\mu_{\mathcal{F}}([1 + \epsilon, +\infty]) = 0$.*

*Proof.* We know from Proposition D.1 that

$$\mu_{\mathcal{F}}(\mathbb{R}) = \sum_{j=1}^{v} j^{-2\alpha - 2\beta} \mu_{\mathcal{F}}^{(j)}(\mathbb{R}) = \sum_{j=1}^{v} j^{-2\alpha - 2\beta} < \infty$$

since $2\alpha + 2\beta > 1$. Finally, the last claim is a direct consequence from Lemma D.1. $\square$

**Proposition D.3** (Point mass at 0). *Let $\alpha > 0$. Suppose $2\alpha + 2\beta > 1$. Then $\mu_{\mathcal{F}}$ has an atom at 0 and there exists some $C > 0$ such that*

$$\left| \mu_{\mathcal{F}}(\{0\}) - \sum_{j=1}^{v} \frac{j^{-2\alpha - 2\beta}}{1 + j^{-2\alpha} d^{2\alpha} \kappa(v/d)} \right| \leq C d^{-2\alpha + (2\beta - 1)_+ - 1}$$

*where $\kappa(v/d)$ solves $\int_0^{v/d} \frac{\kappa}{\kappa + u^{2\alpha}} \, du = 1$.*

*In particular, we know $\mu_{\mathcal{F}}(\{0\}) \asymp d^{-2\alpha + (2\beta - 1)_+}$.*

*Proof.* This is a direct consequence of [80, Prop. E.3, F.1, H.3] and the weak convergence of the Poisson kernels to the Dirac. $\qquad\square$

**Proposition D.4** (Upper-bound for small $\sigma$'s). *Let $\alpha > 0$. There exists some constant $c(\alpha) > 0$ independent of $d$ such that*

$$\forall x \in (0, cd^{-2\alpha}), \quad \mu_{\mathcal{F}}(\mathrm{d}x) = 0.$$

*Proof.* We know that $\mu_{\mathcal{F}}(\mathrm{d}x) = \lim_{\varepsilon \downarrow 0} \frac{1}{\pi} \mathrm{Im}(\langle \mathcal{R}(x + i\varepsilon), (D^{1/2}b)^{\otimes 2}\rangle)\, \mathrm{d}x$. Using [80, Prop. E.3], we know the existence of $c(\alpha) > 0$ such that $m(z)$ is analytic in $\mathcal{B}(0, c(\alpha)d^{-2\alpha})$ and

$$\forall x \in (0, cd^{-2\alpha}), \quad m(x) < 0$$

which implies the result. $\qquad\square$

**Proposition D.5** (Upper-bound for intermediary $\sigma$'s). *Let $\alpha > 0$ with $\alpha \neq \frac{1}{4}$, $\alpha \neq \frac{1}{2}$, and $\beta \neq \frac{1}{2}$ with $2\alpha + 2\beta > 1$, $2\alpha + \beta \neq \frac{1}{2}$ and $2\alpha + 1 > \beta$. There exists $M, \tilde{C} > 0$ depending only on $\alpha, \beta$ such that for any $C \in [d^{-2\alpha}M, \frac{1}{M}]$,*

$$\mu_{\mathcal{F}}([C, 2C]) \leq \tilde{C}\left(C^{1 + \frac{2\beta - 1}{2\alpha}} + \frac{c_\beta}{d}C^{1 - \frac{1}{2\alpha}}\right).$$

*Proof.* Using lemma D.2, we only need to estimate $\sum_{j=1}^{v} j^{-2\alpha - 2\beta}\frac{1}{\pi}\mathrm{Im}\left(\frac{1}{j^{-2\alpha}m(z) - z}\right)$ where $z = u + i\eta(u)$. Applying [80, Prop. E.5], we get the existence of $\epsilon, c, M > 0$ such that for any $u \in [d^{-2\alpha}M, \frac{1}{M}]$, and with $\eta(u) = (\log(1/\epsilon)/c)\max\{u^{1+1/(2\alpha)}, \frac{\pi}{2\alpha}\frac{u^{1-1/(2\alpha)}}{d}\}$ we have

$$m(z(u)) = (1 + \delta_1) + i(1 + \delta_2)\frac{\pi}{2\alpha}u^{-1/(2\alpha)}d^{-1} \quad \text{with} \quad (\delta_1, \delta_2) \in \left[-\frac{1}{3}, \frac{1}{3}\right]^2.$$

Applying the summation lemma [80, Prop. E.4] we obtain for $u \in [d^{-2\alpha}M, \frac{1}{M}]$ and some constants $\bar{\bar{C}}, \bar{\bar{\bar{C}}}$ of $\alpha, \beta, c, \epsilon, M$,

$$\sum_{j=1}^{v} j^{-2\alpha - 2\beta}\frac{1}{\pi}\mathrm{Im}\left(\frac{1}{j^{-2\alpha}m(z(u)) - z(u)}\right) \leq \bar{\bar{C}}\left(u^{\frac{2\beta - 1}{2\alpha}} + \frac{c_\beta}{d}u^{-\frac{1}{2\alpha}} + c_\beta\eta(u)u\log(1/u)\right)$$

$$\leq \bar{\bar{\bar{C}}}\left(u^{\frac{2\beta - 1}{2\alpha}} + \frac{c_\beta}{d}u^{-\frac{1}{2\alpha}}\right).$$

Here, we used $2\alpha + 1 > \beta$ to get $\eta(u)u\log(\frac{1}{u}) \lesssim u^{\frac{2\beta - 1}{2\alpha}}$.

We conclude by writing

$$\begin{aligned}
\mu_{\mathcal{F}}([C, 2C]) &= \int_{\mathbb{R}_+} \mathbb{1}_{[C, 2C]}(s)\mu_{\mathcal{F}}(\mathrm{d}s) \\
&\leq \bar{C}\int_{\mathbb{R}_+}\left(\int_C^{2C} \mathrm{Pois}_{\eta(u)}(s - u)\,\mathrm{d}u\right)\mu_{\mathcal{F}}(\mathrm{d}s) \\
&= \bar{C}\int_C^{2C}\left(\mathrm{Pois}_{\eta(u)} *\mu_{\mathcal{F}}\right)(\mathrm{d}u) \\
&\leq \bar{C}\bar{\bar{\bar{C}}}\int_C^{2C}\left(u^{\frac{2\beta - 1}{2\alpha}} + \frac{c_\beta}{d}u^{-\frac{1}{2\alpha}}\right)\mathrm{d}u \\
&\leq \tilde{C}\left(C^{1 + \frac{2\beta - 1}{2\alpha}} + \frac{c_\beta}{d}C^{1 - \frac{1}{2\alpha}}\right).
\end{aligned}$$

$\qquad\square$

There is some intermediate $\sigma$'s that still need to be bounded, $\sigma^2 \in [c(\alpha)d^{-2\alpha}, Md^{-2\alpha}]$. This is done in the next proposition.

**Proposition D.6.** *Let $\alpha > 0$ with $2\alpha + 2\beta > 1$. Then for any $c_2 > c_1 > 0$ there exists a constant* $C(\alpha, \beta, c_1, c_2)$ *such that*

$$\mu_{\mathcal{F}}([c_1 d^{-2\alpha}, c_2 d^{-2\alpha}]) \leq C(\alpha, \beta, c_1, c_2) d^{-2\alpha + (1-2\beta)_+}.$$

*Proof.* We use the estimate in [80, Prop. E.3] using points $z \in ([c_1, c_2] + i) d^{-2\alpha}$ and the upper-bound $\mathbb{1}_{[c_1, c_2]}(x) \leq \tilde{C} \int_{c_1}^{c_2} \mathrm{Pois}_1(x-s) \, \mathrm{d}s$ for some $\tilde{C}(c_1, c_2) > 0$ and all $x \in \mathbb{R}_+$. $\qquad \square$

### D.1.2 Lower bound on $\mu_{\mathcal{F}}$

To obtain a lower-bound, we will use the lower-bound on the indicator $\mathbb{1}_{[C,2C]}$ using Poisson kernels proved in Lemma D.2. This introduces some error terms due to the tail of the Poisson kernels that need to be controlled using the previous upper-bounds. In particular, we will need $\eta(u) \ll u$, for this bound to be non-vacuous.

**Proposition D.7** (Lower bound for intermediary $\sigma$'s)**.** *Let $\alpha, \beta > 0$, $\alpha > \frac{1}{4}$, $\alpha \neq \frac{1}{2}$, $2\alpha + 2\beta > 1$, $\alpha + 1 > \beta$ there exists some $M > 0$ large enough and $\tilde{C}(\alpha, M)$ such that for any $C \in [Md^{-2\alpha}, \frac{1}{M}]$,*

$$\mu_{\mathcal{F}}([C, 2C]) \geq \frac{1}{\tilde{C}} \left( C^{1 + \frac{2\beta - 1}{2\alpha}} + \frac{c_\beta}{d} C^{1 - \frac{1}{2\alpha}} \right)$$

*Proof.* We consider as in Lemma D.2 $P(u) \stackrel{\mathrm{def}}{=} \int_{x=4C/3}^{5C/3} \mathrm{Pois}_{\eta(x)}(u-x) \, \mathrm{d}x$ where $x \in [d^{-2\alpha} M, \frac{1}{M}]$ and $\eta(x) = (\log(1/\epsilon)/c) \max\{x^{1+1/(2\alpha)}, \frac{\pi}{2\alpha} \frac{x^{1-1/(2\alpha)}}{d}\}$ with $\epsilon, c, M$ given by [80, Prop. E.5]. We write using some constant $\tilde{C}$ from Lemma D.2

$$\tilde{C} \mu_{\mathcal{F}}([C, 2C]) \geq \int_{x=4C/3}^{5C/3} \frac{1}{\pi} \mathrm{Im}((x + i\eta)(-v + (1 - m(x + i\eta))d)) - \int_{u \notin [C, 2C]} P(u) \mu_{\mathcal{F}}(\mathrm{d}u).$$

We hence have using some constant $\tilde{C}_1$ that

$$\tilde{C} \mu_{\mathcal{F}}([C, 2C]) \geq \int_{x=4C/3}^{5C/3} \frac{1}{\pi} \mathrm{Im} \left( \sum_{j=1}^{v} \frac{j^{-2\alpha - 2\beta}}{j^{-2\alpha} m(x + i\eta) - (x + i\eta)} \right)$$

$$- \int_{u \notin [C, 2C]} P(u) \mu_{\mathcal{F}}(\mathrm{d}u)$$

$$\geq \frac{C^{1 + \frac{2\beta - 1}{2\alpha}} + \frac{c_\beta}{d} C^{1 - \frac{1}{2\alpha}}}{\tilde{C}_1} - \left( \int_{u=0}^{C} P(u) \mu_{\mathcal{F}}(\mathrm{d}u) \right.$$

$$\left. + \int_{u=2C}^{1} P(u) \mu_{\mathcal{F}}(\mathrm{d}u) \right).$$

We need to upper-bound both integrals on the right-hand side using Proposition D.5. For the first one we decompose

$$\int_{u=0}^{C} P(u) \mu_{\mathcal{F}}(\mathrm{d}u) \leq \sum_{i=1}^{\infty} \left( \max_{u \in [2^{-i-1}C, 2^{-i}C]} P(u) \times \int_{2^{-i-1}C}^{2^{-i}C} \mu_{\mathcal{F}}(\mathrm{d}u) \right)$$

$$\lesssim \sum_{j=1}^{\log(d^{2\alpha}C)} \frac{\eta(C)C}{C^2} \left( (2^{-i}C)^{1 + \frac{2\beta - 1}{2\alpha}} + \frac{1}{d} (2^{-i}C)^{1 - \frac{1}{2\alpha}} \right)$$

$$\lesssim \frac{\eta(C)}{C} \times \left( C^{1 + \frac{2\beta - 1}{2\alpha}} + \begin{cases} \frac{1}{d} C^{1 - \frac{1}{2\alpha}} & \text{if } 2\alpha > 1 \\ d^{-2\alpha} & \text{if } 2\alpha < 1 \end{cases} \right).$$

Here, we used Proposition D.6 for the left edge and, in the second line $1 + \frac{2\beta-1}{2\alpha} > 1$. There is left to check when $2\alpha < 1$ that

$$\frac{\eta(C)}{C}d^{-2\alpha} \le C^{1+\frac{2\beta-1}{2\alpha}} + \frac{1}{d}C^{1-\frac{1}{2\alpha}}$$

$$\iff \begin{cases} C \ge d^{-\alpha} \text{ and } C^{1/(2\alpha)}d^{-2\alpha} \le C^{1+\frac{2\beta-1}{2\alpha}} + d^{-1}C^{1-1/(2\alpha)} \\ C \le d^{-\alpha} \text{ and } \frac{C^{-1/(2\alpha)}}{d}d^{-2\alpha} \le C^{1+\frac{2\beta-1}{2\alpha}} + d^{-1}C^{1-1/(2\alpha)} \end{cases}.$$

The second case is always valid but the first case is true in particular when $\beta < \alpha + 1$. For the second integral, we similarly expand

$$\int_{2C}^{1} P(u)\mu_{\mathcal{F}}(\mathrm{d}u) \lesssim \sum_{i=1}^{\log(1/C)} \left(\max_{u\in[2^iC,2^{i+1}C]} P(u)\right) \times \int_{2^{-i-1}C}^{2^{-i}C} \mu_{\mathcal{F}}(\mathrm{d}u)$$

$$\lesssim \sum_{j=1}^{\log(\frac{1}{C})} \frac{\eta(C)C}{(2^iC)^2}\left((2^iC)^{1+\frac{2\beta-1}{2\alpha}} + \frac{1}{d}(2^iC)^{1-\frac{1}{2\alpha}}\right)$$

$$\lesssim \frac{\eta(C)}{C}\left(C^{1+\frac{2\beta-1}{2\alpha}} + \frac{1}{d}C^{1-\frac{1}{2\alpha}}\right).$$

Here we used Proposition D.2 for the right edge of the integral, $\beta < \frac{1}{2} + \alpha$ for the first term and $\alpha > 0$ for the second term. In fact for the first term, we can reduce to the less restrictive condition $\beta < 1 + \alpha$ by noticing that $\frac{\eta(C)}{C} \lesssim \left(C^{1+\frac{2\beta-1}{2\alpha}} + \frac{1}{d}C^{1-\frac{1}{2\alpha}}\right)$ under this condition. $\qquad\square$

## D.2 Estimating $\mu_{\mathcal{K}}$

Similarly to the previous sections, we will now upper and lower bound $\mu_{\mathcal{K}}$ to prove in Proposition D.14 upper and lower bounds for the integral of functions with respect to $\mu_{\mathcal{K}}$.

We can rewrite, using [80, Lemma E.1] and dropping the $-xv$ term which vanishes

$$\mu_{\mathcal{K}}(\mathrm{d}x) \overset{\text{def}}{=} \frac{1}{\pi}\lim_{\epsilon\downarrow 0}\mathrm{Im}\left((x+i\epsilon)(1-m(x+i\epsilon))d\right) \quad \text{equivalently} \quad \int_0^\infty \frac{\mu_{\mathcal{K}}(\mathrm{d}s)}{s-z} = z(1-m(z))d.$$
(65)

An important equivalent identity is to rewrite for $z = x + i\eta(x) \in \mathbb{H}$,

$$\frac{1}{\pi}\mathrm{Im}(z((1-m(z))d)) = \int_0^\infty \frac{\mu_{\mathcal{K}}(\mathrm{d}s)\eta(x)}{(s-x)^2+\eta(x)^2}$$
$$= \left(\mathrm{Pois}_{\eta(x)}*\mu_{\mathcal{K}}\right)(x).$$

### D.2.1 Upper-bound on $\mu_{\mathcal{K}}$

Again we decompose the real line into various ranges of $\sigma$'s. Using the upper-bound of the indicator function using Poisson kernels, we similarly get:

**Proposition D.8** (Upper bound for intermediary $\sigma$'s)**.** *Let* $\alpha > 0$, $\alpha \ne \frac{1}{4}$, $\alpha \ne \frac{1}{2}$. *There exists* $M, \tilde{C}$ *depending only on* $\alpha$ *such that for any* $C \in [d^{-2\alpha}M, \frac{1}{M}]$, *we have:*

$$\mu_{\mathcal{K}}([C, 2C]) \le \tilde{C} \times C^{2-\frac{1}{2\alpha}}.$$

*Proof.* Using Lemma D.2, we only need to estimate $\frac{1}{\pi}\mathrm{Im}((x+i\eta)(1-m(x+i\eta))d)$ for $x \in [d^{-2\alpha}M, \frac{1}{M}]$ and $\eta(x) = (\log(1/\epsilon)/c)\max\{x^{1+1/(2\alpha)}, \frac{\pi}{2\alpha}\frac{x^{1-1/(2\alpha)}}{d}\}$. Applying [80, Prop E.5], there exists $c, M > 0$ and $\epsilon > 0$ sufficiently small such that

$$\frac{1}{\pi}\mathrm{Im}((x+i\eta(x))(1-m(x+i\eta(x)))d) = \frac{x^{1-1/(2\alpha)}}{2\alpha} + \mathcal{O}\left(\eta(x)\mathcal{A}(x+i\eta(x)) + \epsilon x^{1-1/(2\alpha)}\right).$$

We use the bound on $\mathcal{A}$ from [80, Prop.E.4] (when setting $\beta = 0$) to conclude. $\qquad\square$

**Proposition D.9** (Upper bound for large $\sigma$'s)**.** *If $\alpha > \frac{1}{4}$, $\alpha \neq \frac{1}{2}$, $\forall M > 0$, $\mu_{\mathcal{K}}([\frac{1}{M}, M]) \lesssim 1$. Additionally, for any $\epsilon > 0$, there exists some $\overline{d} > 0$ such that $\forall d \geq \overline{d}$, $\mu_{\mathcal{K}}([1 + \epsilon, +\infty)) = 0$.*

*Proof.* Consider $z = x + i \in \mathbb{H}$ for $x \in [\frac{1}{M}, M]$. This defines a compact $U$ of distance at least 1 to $[0, 1]$. From [80, Prop E.6], we have that

$$
\begin{aligned}
\text{Im}(z(1 - m(z))d) &= \text{Im}\left( z \sum_{j=1}^{v} \frac{j^{-2\alpha}}{j^{-2\alpha} - z} \right) + \mathcal{O}\left( \frac{C(\alpha)}{1 \times \min\{d, d^{4\alpha-1}\}} \right) \\
&= x \times \sum_{j=1}^{v} \frac{j^{-2\alpha} \times 1}{((j^{-2\alpha} - x)^2 + 1} + 1 \times \sum_{j=1}^{v} \frac{j^{-2\alpha} \times (j^{-2\alpha} - x)}{((j^{-2\alpha} - x)^2 + 1} \\
&\quad + \mathcal{O}\left( \frac{C(\alpha)}{1 \times \min\{d, d^{4\alpha-1}\}} \right) \\
&= \sum_{j=1}^{v} \frac{j^{-4\alpha}}{(j^{-2\alpha} - a)^2 + 1} + \mathcal{O}\left( \frac{C(\alpha)}{1 \times \min\{d, d^{4\alpha-1}\}} \right).
\end{aligned}
$$

The above being bounded, it implies using Lemma D.2 that $\mu_{\mathcal{K}}([\frac{1}{M}, M]) \lesssim 1$. Finally, the last claim is a direct consequence of Lemma D.1. $\qquad\square$

**Proposition D.10** (Upper bound for $\sigma$'s near zero)**.** $\exists c(\alpha) > 0$ *such that* $\mu_{\mathcal{K}}([0, c(\alpha)d^{-2\alpha}]) = 0$.

*Proof.* Using [80, Prop. E.3], we know that $m(z)$ is analytic in $\mathcal{B}(0, c(\alpha)d^{-2\alpha})$. Hence, from (65) it follows that $\mu_{\mathcal{K}}$ is null on $[0, c(\alpha)d^{-2\alpha}]$. $\qquad\square$

We now move to the last range for $\sigma$'s:

**Proposition D.11.** *Let $\alpha, \beta > 0$ with $2\alpha + 2\beta > 1$. Then $\forall c_2 > c_1 > 0$, we have $\mu_{\mathcal{K}}([c_1 d^{-2\alpha}, c_2 d^{-2\alpha}]) \lesssim d^{1-4\alpha}$.*

*Proof.* We use the estimate in [80, Prop. E.3] using points $z \in ([c_1, c_2] + i)d^{-2\alpha}$ and the upper-bound $\mathbb{1}_{[c_1, c_2]}(x) \leq \tilde{C} \int_{c_1}^{c_2} \text{Pois}_{s,1}(x) \, ds$. This brings that

$$
\begin{aligned}
\mu_{\mathcal{K}}([c_1 d^{-2\alpha}, c_2 d^{-2\alpha}]) &\lesssim \int_{c_1 d^{-2\alpha}}^{c_2 d^{-2\alpha}} (\text{Pois}_1 * \mu_{\mathcal{K}})(s) \, ds \\
&\lesssim \int_{c_1}^{c_2} d^{-4\alpha} \, \text{Im}\left( (s+i)(1 - m(s+i))d \right) \\
&= d^{1-4\alpha} \int_{c_1}^{c_2} \text{Im}\left( (s+i)(1 - \mathfrak{f}(s+i)) \right) + \mathcal{O}(d^{-4\alpha}) \\
&\lesssim d^{1-4\alpha}.
\end{aligned}
$$

$\qquad\square$

### D.2.2 Lower bound on $\mu_{\mathcal{K}}$

In the next proposition, we prove a lower bound on $\mu_{\mathcal{K}}$.

**Proposition D.12** (Lower bound for intermediary $\sigma$'s)**.** *Let $\alpha > \frac{1}{4}, \alpha \neq \frac{1}{2}$ there exists some $M > 0$ large enough and $\tilde{C}(\alpha, M)$ such that for any $C \in [Md^{-2\alpha}, \frac{1}{M}]$,*

$$
\mu_{\mathcal{K}}([C, 2C]) \geq \frac{C^{2 - 1/(2\alpha)}}{\tilde{C}}.
$$

*Proof.* We consider as in Lemma D.2 $P(u) \stackrel{\text{def}}{=} \int_{x=4C/3}^{5C/3} \mathrm{Pois}_{\eta(x)}(u - x)\,\mathrm{d}x$. Again for $x \in [d^{-2\alpha}M, \frac{1}{M}]$ and $\eta(x) = (\log(1/\epsilon)/c)\max\{x^{1+1/(2\alpha)}, \frac{\pi}{2\alpha}\frac{x^{1-1/(2\alpha)}}{d}\}$ with $\epsilon, c, M$ given by [80, Prop.E.5]

We write with some constant $\tilde{C}$ from Lemma D.2 and potentially larger $\bar{C}$

$$\tilde{C}\mu_{\mathcal{K}}([C, 2C]) \geq \int_{x=4C/3}^{5C/3} \frac{1}{\pi} \mathrm{Im}((x + i\eta)(-v + (1 - m(x + i\eta))d))\,\mathrm{d}x - \int_{u \notin [C,2C]} P(u)\mu_{\mathcal{K}}(\mathrm{d}u)$$

$$\geq \frac{C^{2-1/(2\alpha)}}{\bar{C}} - \left( \int_{u=0}^{C} P(u)\mu_{\mathcal{K}}(\mathrm{d}u) + \int_{u=2C}^{1} P(u)\mu_{\mathcal{K}}(\mathrm{d}u) \right).$$

We need to upper-bound both integrals on the right-hand side. For that we use Proposition D.8. For the first one we decompose

$$\int_{u=0}^{C} P(u)\mu_{\mathcal{K}_{pp}}(\mathrm{d}u) \leq \sum_{i=1}^{\infty} \left( \max_{u \in [2^{-i-1}C, 2^{-i}C]} P(u) \right) \times \int_{2^{-i-1}C}^{2^{-i}C} \mu_{\mathcal{K}}(\mathrm{d}u)$$

$$\lesssim \sum_{i=1}^{\infty} \frac{\eta(C)C}{(C/3)^2} \times \left(2^{-i}C\right)^{2-\frac{1}{2\alpha}}$$

$$\lesssim \frac{\eta(C)}{C}C^{2-\frac{1}{2\alpha}}$$

$$\lesssim M^{-1/(2\alpha)}C^{2-\frac{1}{2\alpha}}.$$

Here we used that $2 - \frac{1}{2\alpha} > 0$ for the sum to converge. For the second one we write

$$\int_{u=2C}^{1} P(u)\mu_{\mathcal{K}}(\mathrm{d}u) \lesssim \sum_{i=1}^{\log(1/C)} \left( \max_{u \in [2^iC, 2^{i+1}C]} P(u) \right) \times \int_{2^iC}^{2^{i+1}C} \mu_{\mathcal{K}}(\mathrm{d}u)$$

$$\lesssim \sum_{i=1}^{\infty} \frac{\eta(C)C}{(C2^i)^2} \times \left(2^iC\right)^{2-\frac{1}{2\alpha}}$$

$$\lesssim \frac{\eta(C)}{C}C^{2-\frac{1}{2\alpha}}$$

$$\lesssim M^{-1/(2\alpha)}C^{2-\frac{1}{2\alpha}}.$$

All this finally brings that for $M$ large enough, $\mu_{\mathcal{K}}([C, 2C]) \geq \frac{1}{\tilde{C}}C^{2-\frac{1}{2\alpha}}$ for some $\tilde{C}$ depending on $\epsilon, c, \alpha$.

$\square$

## D.3  Forcing function

In this section, we decompose the measure $\mu_{\mathcal{F}}$ based on the upper and lower bounds previously derived. This *decomposition* allows us to break the forcing function into three components, $\mathcal{F}_0(t)$, $\mathcal{F}_{pp}(t)$, and $\mathcal{F}_{ac}(t)$, that is

$$\mathcal{F}(t) \asymp \mathcal{F}_{pp}(t) + \mathcal{F}_{ac}(t) + \mathcal{F}_0(t).$$

We will discuss this in detail.

Proposition D.3 gives an explicit formula for the behavior of the measure $\mu_{\mathcal{F}}$ at $x = 0$. Proposition D.4 gives the "gap" between the $0$ eigenvalues of $\hat{K}$ and the next smallest eigenvalue which occurs at $c(\alpha) \times d^{-2\alpha}$ where $c(\alpha)$ is some explicit constant. Moreover Proposition D.2 says the largest eigenvalue of $\hat{K}$ is approximately $1$. This is all to say, the support of $\mu_{\mathcal{F}}$ is $\{0\} \cup [c(\alpha)d^{-2\alpha}, 1 + c(\alpha)]$. Lastly Propositions D.5 and D.7 can be interpreted as saying something about the density for $\mu_{\mathcal{F}}$.

$$\mathcal{P}(t) \asymp \hat{\mathcal{F}}_{pp}(\vartheta(t)) + \hat{\mathcal{F}}_{ac}(\vartheta(t)) + \mathcal{F}_0(t) + \gamma\hat{\mathcal{K}}_{pp}(\vartheta(t))$$
$$\asymp \mathcal{F}_{pp}(t) + \mathcal{F}_{ac}(t) + \mathcal{F}_0(t) + \tfrac{1}{\gamma B}\mathcal{K}_{pp}(t)$$

| | |
|---|---|
| **Model Capacity:** $\mu_{\mathcal{F}}(\{0\})$ | $\mathcal{F}_0(t) \asymp d^{-2\alpha+\max\{0,1-2\beta\}}$ |
| **Population Bias:** Spikes in $\mu_{\mathcal{F}}$ | $\hat{\mathcal{F}}_{pp}(t) \asymp t^{-(2\alpha+2\beta-1)/(2\alpha)}$ |
| **Embedding Bias:** Bulk of $\mu_{\mathcal{F}}$ | $\hat{\mathcal{F}}_{ac}(t) \leq \begin{cases} C \times \mathcal{F}_0(t), & \text{if } 2\beta > 1, 2\alpha < 1 \\ 0, & \text{if } 2\beta < 1 \end{cases}$ |
| | and if $2\beta > 1, 2\alpha > 1$, $\hat{\mathcal{F}}_{ac}(t) \asymp d^{-1}t^{-1+1/(2\alpha)}$ |
| **Variance:** Spikes of $\mu_{\mathcal{K}}$ | $\hat{\mathcal{K}}_{pp}(t) \asymp d^{\max\{0,1-2\alpha\}}t^{-2+1/(2\alpha)}$ |

| Algorithm | Learning rate $\gamma$ | Time scale, $\vartheta(t)$ |
|---|---|---|
| SGD($\gamma_2$) | $\gamma_2$ | $\vartheta(t) = 1 + \gamma_2 Bt$ |
| SGD-M($\gamma_2, \gamma_3, \delta$) | $\gamma_2 + \frac{\gamma_3}{\delta}$ | $\vartheta(t) = 1 + (\gamma_2 + \frac{\gamma_3}{\delta})Bt$ |
| DANA-constant($\gamma_2, \gamma_3$)$^\dagger$, $t \leq d$ | $\gamma_2$ | $\vartheta(t) = 1 + \gamma_2 Bt$ |
| DANA-constant($\gamma_2, \gamma_3$)$^\dagger$, $t \geq d$ | $\gamma_2$ | $\vartheta(t) = 1 + \gamma_3 Bt^2$ |
| DANA-decaying($\gamma_2, \gamma_3$)$^\#$, $\gamma_3(t) \asymp (1+t)^{-1/(2\alpha)}$ | $\gamma_2$ | $\vartheta(t) = 1 + B(1+t)^{2-1/(2\alpha)}$ |
| DANA($\gamma_2, \gamma_3$), $\gamma_3(t; d)$ general schedule | $\gamma_2$ | $\vartheta(t) = 1 + \gamma_2 Bt + \left(\int_0^t \sqrt{\gamma_3(s;d)B}\, \mathrm{d}s\right)^2$ |

$^\dagger$ DANA-constant with $\gamma_3 \asymp \gamma_2 \times 1/d$.   $^\#$ DANA-decaying only when $2\alpha > 1$ and $\gamma_2 \asymp 1$.

Table 4: **Asymptotics for the forcing and kernel functions for all algorithms.** See Section G/Section F for details/proofs of the derivations for these asymptotics of SGD-M/SGD. See Section H for details/proofs of the asymptotics for DANA-constant. See Section I for details about the heuristics used to derive these asymptotics for DANA-decaying. Here the constant $C$ is independent of dimension and $B$ is order 1 independent of $d$. For the DANA class, we do not have a full proof. We believe this is true for $2\alpha > 1$ and $\tilde{\gamma}_3 d^{-\kappa_3} \leq \gamma_2$, which we believe is true for stability reasons and would suggest that $\gamma = \gamma_2$; this is the most uncertain part. Note that $\hat{\mathcal{K}}_{pp} \circ \vartheta = \frac{1}{\gamma^2 B}\mathcal{K}_{pp}$ where $\hat{\mathcal{K}}_{pp}$ is defined in Thm. 4.1 and $\mathcal{K}_{pp}(t) \stackrel{\text{def}}{=} \mathcal{K}_{pp}(t, 0)$ is defined in this section.

Therefore, informally, we have
$$\mu_{\mathcal{F}} \approx \mu_{\mathcal{F}_{pp}} + \mu_{\mathcal{F}_{ac}} + \mu_{\mathcal{F}_0}, \tag{66}$$
where we define the three measures as
$$\mu_{\mathcal{F}_0}(\mathrm{d}u) \stackrel{\text{def}}{=} \mu_{\mathcal{F}}(\{0\})\delta_0(\mathrm{d}u), \quad \mu_{\mathcal{F}_{pp}}(\mathrm{d}u) \stackrel{\text{def}}{=} \mathbb{1}_{0<u\leq 1} u^{(2\beta-1)/(2\alpha)}\,\mathrm{d}u,$$
$$\text{and} \quad \mu_{\mathcal{F}_{ac}}(\mathrm{d}u) \stackrel{\text{def}}{=} \mathbb{1}_{d^{-2\alpha}<u\leq 1} \frac{c_\beta u^{-1/(2\alpha)}}{d}\,\mathrm{d}u.$$
Here $c_\beta = \sum_{j=1}^\infty j^{-2\beta}$ if $\beta > \frac{1}{2}$ and 0 otherwise and $\delta_0$ is a Dirac delta function, that is,
$$\delta_0(x) \stackrel{\text{def}}{=} \begin{cases} 0, & x \neq 0 \\ \infty, & x = 0, \end{cases} \quad \text{where} \quad \int_{-\infty}^\infty \delta_0(x)\,\mathrm{d}x = 1.$$
With this interpretation, we decompose the forcing function based on integrating against the three different measures $\mu_{\mathcal{F}_0}, \mu_{\mathcal{F}_{pp}}, \mu_{\mathcal{F}_{ac}}$, that is,
$$\mathcal{F}_0(t) \stackrel{\text{def}}{=} \int_0^\infty (\Phi_u(t,0))_{11} \times \mu_{\mathcal{F}_0}(\mathrm{d}u) = \mu_{\mathcal{F}_0}(\{0\})(\Phi_0(t,0))_{11},$$
$$\mathcal{F}_{pp}(t) \stackrel{\text{def}}{=} \int_0^\infty (\Phi_u(t,0))_{11} \times \mu_{\mathcal{F}_{pp}}(\mathrm{d}u) = \int_0^1 (\Phi_u(t,0))_{11} \times u^{(2\beta-1)/(2\alpha)}\,\mathrm{d}u, \tag{67}$$
$$\text{and} \quad \mathcal{F}_{ac}(t) \stackrel{\text{def}}{=} \int_0^\infty (\Phi_u(t,0))_{11} \times \mu_{\mathcal{F}_{ac}}(\mathrm{d}u) = \frac{c_\beta}{d}\int_{d^{-2\alpha}}^1 (\Phi_u(t,0))_{11} \times u^{-1/(2\alpha)}\,\mathrm{d}u.$$

In particular, for any "nice" function $\Phi_u(t) : \mathbb{R}_{\geq 0} \times \mathbb{R}_{\geq 0} \to \mathbb{R}$, with some regularity parameter $\Lambda$ we know the existence of a constant $C(\alpha, \beta, \Lambda)$ such that $\forall t \geq 0$:

$$\frac{1}{C} \times (\mathcal{F}_0(t) + \mathcal{F}_{pp}(t) + \mathcal{F}_{ac}(t)) \leq \mathcal{F}(t) \leq C \times (\mathcal{F}_0(t) + \mathcal{F}_{pp}(t) + \mathcal{F}_{ac}(t)).$$

We formalize this idea in the next proposition.

**Proposition D.13.** *Let* $\alpha > \frac{1}{4}, \beta > 0$ *with* $2\alpha + 2\beta > 1$, $\alpha + 1 > \beta$, $\alpha, \beta \neq \frac{1}{2}$, $\Lambda_1 > 0$. *There exists* $M, M_1, M_2 > 0$ *and* $C(\alpha, \Lambda_1)$ *such that for any* $f : [0,1] \to \mathbb{R}_+$ *satisfying* $\forall u \in (0, \frac{1}{4})$

$$\min_{[u,2u]} f(u) \geq \Lambda_1 \max_{[2u,4u]} f(u).$$

*we have for any* $d \geq 1$,

$$\frac{1}{C(\alpha, \Lambda_1)} \int_{M_1 d^{-2\alpha}}^{\frac{1}{M_1}} f(u) \left( \mu_{\mathcal{F}_{pp}} + \mu_{\mathcal{F}_{ac}} \right)(du) \leq \int_{Md^{-2\alpha}}^{\frac{1}{M}} f(u) \mu_{\mathcal{F}}(du)$$

$$\leq C(\alpha, \Lambda_1) \int_{M_2 d^{-2\alpha}}^{\frac{1}{M_2}} f(u) \left( \mu_{\mathcal{F}_{pp}} + \mu_{\mathcal{F}_{ac}} \right)(du).$$

*Proof.* The proof is entirely similar to the one of Proposition D.14 $\qquad\square$

**Remark D.1.** *An alternative (although not identical) definition for* $\mu_{\mathcal{F}_{pp}}$ *is as a sum of point-masses*

$$\mu_{\mathcal{F}_{pp}}(du) = \sum_{j=1}^{v} j^{-(2\beta+2\alpha)} \delta_{j^{-2\alpha}}(du).$$

## D.4 Kernel function

We can simplify the kernel function in a similar way as for the forcing function, that is

$$\mathcal{K}(t,s) \overset{\text{def}}{=} \mathcal{K}_s(t) \asymp \mathcal{K}_{pp}(t,s),$$

where $\mathcal{K}_{pp}$ that is simpler to analyze than directly the kernel function.

Two differences are that there is no mass at $0$ and the absolutely-continuous part is negligible. Hence only the pure-point contribution survives. We define accordingly

$$\mu_{\mathcal{K}_{pp}}(du) \overset{\text{def}}{=} \mathbb{1}_{0 < u \leq 1} u^{1 - 1/(2\alpha)} \, du \quad \text{where} \quad \mu_{\mathcal{K}} \approx \mu_{\mathcal{K}_{pp}}, \tag{68}$$

and the kernel function integration against this measure $\mu_{\mathcal{K}_{pp}}$ by

$$\mathcal{K}_{pp}(t,s) \overset{\text{def}}{=} B \int_0^\infty \left( \gamma_2^2(s)(\Phi_u(t,s))_{11} + \gamma_1^2(s)(\Phi_u(t,s))_{12} \right) \times \mu_{\mathcal{K}_{pp}}(du)$$

$$= B \int_0^1 \left( \gamma_2^2(s)(\Phi_u(t,s))_{11} + \gamma_1^2(s)(\Phi_u(t,s))_{12} \right) u^{1 - 1/(2\alpha)} \, du. \tag{69}$$

In particular, for $\alpha > \frac{1}{4}$, $\alpha \neq \frac{1}{2}$ and for any "nice" functions $\Phi_u(t,s)_{12}, \Phi_u(t,s)_{12} : \mathbb{R}_{\geq 0} \times \mathbb{R}_{\geq 0} \times \mathbb{R}_{\geq 0} \to \mathbb{R}$, with some regularity parameter $\Lambda$ we know the existence of a constant $C(\alpha, \Lambda)$ such that $\forall t \geq 0$:

$$\frac{1}{C} \times \mathcal{K}_{pp}(t,s) \leq \mathcal{K}(t,s) \leq C \times \mathcal{K}_{pp}(t,s).$$

We formalize this idea in the next proposition.

**Proposition D.14.** *Let* $\alpha > \frac{1}{4}$, $\Lambda_1 > 0$. *There exists* $M, M_1, M_2 > 0$ *and* $C(\alpha, \Lambda_1)$ *such that for any* $f : [0,1] \to \mathbb{R}_+$ *satisfying* $\forall u \in (0, \frac{1}{4})$

$$\min_{[u,2u]} f(u) \geq \Lambda_1 \max_{[2u,4u]} f(u).$$

*we have for any* $d \geq 1$,

$$\frac{1}{C(\alpha, \Lambda_1)} \int_{M_1 d^{-2\alpha}}^{\frac{1}{M_1}} f(u) \mu_{\mathcal{K}_{pp}}(du) \leq \int_{Md^{-2\alpha}}^{\frac{1}{M}} f(u) \mu_{\mathcal{K}}(du) \leq C(\alpha, \Lambda_1) \int_{M_2 d^{-2\alpha}}^{\frac{1}{M_2}} f(u) \mu_{\mathcal{K}_{pp}}(du).$$

*Proof.* We will instead show the following claim, which directly implies the result.

**Claim D.1.** *There exists $\bar{k}(\alpha) \in \mathbb{N}^*$ and $C(\alpha, \Lambda_1)$ such that for any $f : [0,1] \to \mathbb{R}_+$ satisfying $\forall u \in (0, \frac{1}{4})$ the following holds*

$$\min_{[u,2u]} f(u) \geq \Lambda_1 \max_{[2u,4u]} f(u).$$

*then for any $d \geq 1$ and any $k_1, k_2 \in \mathbb{N}_+$ with $2\alpha \log(d) - \bar{k} \geq k_1 \geq k_2 \geq \bar{k}$ we have*

$$\frac{1}{C(\alpha, \Lambda_1)} \int_{2^{-k_1+1}}^{2^{-k_2+1}} f(u)\mu_{\mathcal{K}_{pp}}(\mathrm{d}u) \leq \int_{2^{-k_1}}^{2^{-k_2}} f(u)\mu_{\mathcal{K}}(\mathrm{d}u) \leq C(\alpha, \Lambda_1) \int_{2^{-k_1-1}}^{2^{-k_2-1}} f(u)\mu_{\mathcal{K}_{pp}}(\mathrm{d}u).$$

*Proof of the claim*: First, using Proposition D.8 and Proposition D.12 we know the existence of $\bar{k}(\alpha)$ and $\tilde{C}(\alpha)$ such that for any $2\alpha \log(d) - \bar{k} \geq k_1 \geq k_2 \geq \bar{k}$, and $k \in [k_2 - 1, k_1 + 1]$, $\frac{1}{\tilde{C}}\mu_{\mathcal{K}_{pp}}([2^{-k}, 2^{-k+1}]) \leq \mu_{\mathcal{K}}([2^{-k}, 2^{-k+1}]) \leq \tilde{C}\mu_{\mathcal{K}_{pp}}([2^{-k}, 2^{-k+1}])$.

For the left side we write

$$\int_{2^{-k_1+1}}^{2^{-k_2+1}} f(u)\mu_{\mathcal{K}_{pp}}(\mathrm{d}u) \leq \sum_{k=k_2}^{k_1-1} \left( \max_{u \in [2^{-k+1}, 2^{-k}]} f(u) \right) \times \mu_{\mathcal{K}_{pp}}([2^{-k+1}, 2^{-k}])$$

$$\leq \sum_{k=k_2}^{k_1-1} \left( \frac{1}{\Lambda_1} \min_{u \in [2^{-k}, 2^{-k-1}]} f(u) \right) \times \tilde{C}(\alpha)\mu_{\mathcal{K}}([2^{-k}, 2^{-k-1}])$$

$$\leq \frac{\tilde{C}(\alpha)}{\Lambda_1} \int_{2^{-k_1}}^{2^{-k_2}} f(u)\mu_{\mathcal{K}}(\mathrm{d}u).$$

Similarly for the right side:

$$\int_{2^{-k_1-1}}^{2^{-k_2-1}} f(u)\mu_{\mathcal{K}_{pp}}(\mathrm{d}u) \geq \sum_{k=k_2}^{k_1-1} \left( \min_{u \in [2^{-k-1}, 2^{-k}]} f(u) \right) \times \mu_{\mathcal{K}_{pp}}([2^{-k-1}, 2^{-k}])$$

$$\geq \sum_{k=k_2}^{k_1-1} \left( \Lambda_1 \max_{u \in [2^{-k}, 2^{-k+1}]} f(u) \right) \times \frac{1}{\tilde{C}(\alpha)}\mu_{\mathcal{K}}([2^{-k}, 2^{-k+1}])$$

$$\geq \frac{\Lambda_1}{\tilde{C}(\alpha)} \int_{2^{-k_1}}^{2^{-k_2}} f(u)\mu_{\mathcal{K}}(\mathrm{d}u).$$

Taking $C(\alpha, \Lambda_1) \stackrel{\mathrm{def}}{=} \frac{\tilde{C}(\alpha)}{\Lambda_1}$ yields the result.

$\square$

**Remark D.2.** *An alternative (although not identical) definition for $\mu_{\mathcal{K}_{pp}}$ is as a sum of point-masses*

$$\mu_{\mathcal{K}_{pp}}(\mathrm{d}u) = \sum_{j=1}^{v} j^{-4\alpha}\delta_{j^{-2\alpha}}(\mathrm{d}u).$$

We can finally provide the following heuristics on the different loss terms:

- "Population bias" ($\mathcal{F}_{pp}$): This loss term corresponds to the loss dynamics when running full-batch gradient descent on the problem (hence following the population gradient), without the embedding matrix $W$.
- "Model capacity" ($\mathcal{F}_0$): This loss term (which is only $d$-dependent) represents the limit of the loss, as the number of iterations reaches infinity. It arises from the partial expressivity of our model class, since the learned parameters $\theta \in \mathbb{R}^d$ cannot encode the whole target vector $b \in \mathbb{R}^v$ when $v > d$.

- "Embedding bias" ($\mathcal{F}_{ac}$): This loss term comes from the random embedding matrix $W$ which deforms the spectrum of the data covariance matrix and misaligns it with the target vector $b$.
- "Variance" ($\mathcal{K}_{pp}$): This loss term comes from the stochasticity of the algorithm which at each step samples a new i.i.d. random datapoint to compute a stochastic gradient. This stochastic gradient, whose average recovers the population gradient, is a non-exact estimate of the gradient and therefore introduces this additional loss term.

# E   Compute-optimal curves - General

We summarize the results of this Section E in Figure 12 (Phase Diagrams) and for scaling laws and compute-optimal tradeoffs see Figure 13 and 14 (above the high-dimensional line) and Figure 15 (below the high-dimensional line). In the Phase Diagrams, we indicate which phases acceleration occurs.

In this section for finding the compute-optimal curves, we note that DANA-decaying refers to DANA with $\kappa_3 = \frac{1}{2\alpha}$ and $\kappa_2 = \kappa_1 = \max\{0, 1 - 2\alpha\}$. This is the same as saying that $\gamma_2$ is the largest possible learning rate for stability. DANA-constant refers to DANA with $\kappa_1 = \max\{0, 1 - 2\alpha\}$, $\kappa_2 = 1 + \kappa_1$, and $\kappa_3 = 0$.

In light of (62) and the simplifications of the forcing function (Section D.3) and kernel function (Section D.4), we have that

$$\mathcal{P}(t) \asymp \mathcal{F}_{pp}(t) + \mathcal{F}_{ac}(t) + \mathcal{F}_0(t) + \frac{1}{\gamma B}\mathcal{K}_{pp}(t), \tag{70}$$

where $\gamma = \gamma_2$ for SGD/DANA and $\gamma = \gamma_2 + \frac{\gamma_3}{\delta}$ for SGD-M, and $\mathcal{K}_{pp}(t) \overset{\text{def}}{=} \mathcal{K}_{pp}(t, 0)$.

Since for each algorithm we consider (SGD, SGD-M, DANA-constant, DANA-decaying) these terms are asymptotically equal to $d^{-\tau}t^{-\sigma}$, we can now derive the compute-optimal curves and exponents. See Table 4 for the asymptotics of the forcing and kernel functions for each of these algorithms. For derivations, see Section G (SGD-M), Section H (DANA-constant), and Section I (DANA-decaying).

To simplify the computations for compute-optimal curves, we introduce the following curve

$$\tilde{\mathcal{P}}(t) \overset{\text{def}}{=} \max\left\{\mathcal{F}_{pp}(t), \mathcal{F}_{ac}(t), \mathcal{F}_0(t), \frac{1}{\gamma B}\mathcal{K}_{pp}(t)\right\}. \tag{71}$$

The function $\tilde{\mathcal{P}}(t, d)$ achieves the same power law behavior as the original compute-optimal curve $\mathcal{P}(t, d)$ (i.e., the slope of the compute-optimal curve is correct) and deviates from the true curve by an absolute constant (independent of $d$ and $\mathfrak{f}$). Note that some of the terms in the max function (71) should be taken to be $0$ when not defined for the different phases. Therefore, we derive the compute-optimal curves by solving the problem

$$\min_d \tilde{\mathcal{P}}\left(\frac{\mathfrak{f}}{d \cdot B}, d\right), \quad \text{and if } d^\star(\mathfrak{f}) \overset{\text{def}}{=} \arg\min_d \tilde{\mathcal{P}}\left(\frac{\mathfrak{f}}{d \cdot B}, d\right),$$

$$\text{then the compute-optimal curve is} \quad \tilde{\mathcal{P}}^\star(\mathfrak{f}) \overset{\text{def}}{=} \tilde{\mathcal{P}}\left(\frac{\mathfrak{f}}{d^\star(\mathfrak{f}) \cdot B}, d^\star(\mathfrak{f})\right). \tag{72}$$

Using this alternative loss function, $\tilde{\mathcal{P}}(t, d)$, the compute-optimal line must occur at one of the corner points, i.e., where any pair of functions equal each other. The following lemma gives a useful characterization of these points.

**Lemma E.1** (See Lemma D.1 in [80]). *Suppose* $\mathcal{C}_0, \mathcal{C}_1 > 0$ *are constants and* $\gamma_0, \gamma_1, p_0, p_1 \in \mathbb{R}$ *exponents and such that a function* $\hat{\mathcal{P}}(t, d)$ *equals*

$$\hat{\mathcal{P}}(t, d) = \max\left\{\mathcal{C}_0 t^{-\gamma_0}d^{-p_0}, \mathcal{C}_1 t^{-\gamma_1}d^{-p_1}\right\}.$$

*Suppose that there exists* $i \in \{0, 1\}$, $j \overset{\text{def}}{=} 1 - i$ *such that* $p_i - \gamma_i > 0 > p_j - \gamma_j$. *Then replacing* $r \mapsto \frac{\mathfrak{f}}{d}$ *the minimizer of* $\hat{\mathcal{P}}$ *in* $d$ *satisfies*

$$d^\star \overset{\text{def}}{=} \arg\min_d \{\hat{\mathcal{P}}(\mathfrak{f}, d)\} = \left(\frac{\mathcal{C}_0}{\mathcal{C}_1}\right)^{1/(\gamma_1 - p_1 - \gamma_0 + p_0)} \times \mathfrak{f}^{(-\gamma_0 + \gamma_1)/(\gamma_1 - p_1 - \gamma_0 + p_0)}$$

*and the associated value is*

$$\min_d \hat{\mathcal{P}}(\mathfrak{f}, d) = \mathcal{C}_0 \times \mathfrak{f}^{-\gamma_0} \times (d^\star)^{\gamma_0 - p_0}.$$

| | Loss $\mathcal{P}(t)$ | Trade off | Compute-optimal Curves | | |
|---|---|---|---|---|---|
| **Phase I** | $\mathcal{F}_{pp}(t) + \mathcal{F}_0(t)$ | $\mathcal{F}_{pp} = \mathcal{F}_0$ | **Ia** | $\tilde{\mathcal{P}}^\star_{\text{Phase Ia}}(\mathfrak{f}) \asymp \mathfrak{f}^{\left(\frac{1}{2\alpha+1}-1\right)(1+\beta/\alpha-1/(2\alpha))}$ 
 $d^\star_{\text{Phase Ia}} \asymp \mathfrak{f}^{1/(2\alpha+1)}$ | |
| | | | **Ib** | $\tilde{\mathcal{P}}^\star_{\text{Phase Ib}}(\mathfrak{f}) \asymp \mathfrak{f}^{\frac{1}{2}-\alpha-\beta}$ 
 $d^\star_{\text{Phase Ib}} \asymp \mathfrak{f}^{\frac{1}{2}}$ | |
| | | | **Ic** | $\tilde{\mathcal{P}}^\star_{\text{Phase Ic}}(\mathfrak{f}) \asymp \mathfrak{f}^{\frac{\alpha(2\alpha+2\beta-1)}{\alpha(2\beta-3)-2\beta+1}}$ 
 $d^\star_{\text{Phase Ic}} \asymp \mathfrak{f}^{\frac{1-2(\alpha+\beta)}{2(\alpha(2\beta-3)-2\beta+1)}}$ | |
| **Phase II** | $\mathcal{F}_{pp}(t) + \mathcal{F}_{ac}(t)$ 
 $+\mathcal{F}_0(t)$ | $\mathcal{F}_{pp} = \mathcal{F}_{ac}$ | $\tilde{\mathcal{P}}^\star_{\text{Phase II}}(\mathfrak{f}) \asymp \mathfrak{f}^{-\frac{2\alpha+2\beta-1}{2(\alpha+\beta)}}$ 
 $d^\star_{\text{Phase II}} \asymp \mathfrak{f}^{(\beta/\alpha)/(1+\beta/\alpha)}$ | | |
| **Phase III** | $\mathcal{F}_{ac}(t) + \mathcal{F}_0(t)$ 
 $+\frac{1}{\gamma B}\mathcal{K}_{pp}(t)$ | $\frac{1}{\gamma B}\mathcal{K}_{pp} = \mathcal{F}_{ac}$ | $\tilde{\mathcal{P}}^\star_{\text{Phase III}}(\mathfrak{f}) \asymp \mathfrak{f}^{(1-4\alpha)/(4\alpha)}$ 
 $d^\star_{\text{Phase III}} \asymp \mathfrak{f}^{1/2}$ | | |
| **Phase IV** | $\mathcal{F}_{pp}(t) + \mathcal{F}_0(t)$ 
 $+\frac{1}{\gamma B}\mathcal{K}_{pp}(t)$ | **IVa** $\frac{1}{\gamma B}\mathcal{K}_{pp} = \mathcal{F}_0$ | $\tilde{\mathcal{P}}^\star_{\text{Phase IVa}}(\mathfrak{f}) \asymp \mathfrak{f}^{-\alpha}$ 
 $d^\star_{\text{Phase IVa}} \asymp \mathfrak{f}^{1/2}$ | | |
| | | **IVb** $\frac{1}{\gamma B}\mathcal{K}_{pp} = \mathcal{F}_{pp}$ | $\tilde{\mathcal{P}}^\star_{\text{Phase IVb}}(\mathfrak{f}) \asymp \mathfrak{f}^{\frac{(1-2\alpha)(2\alpha+2\beta-1)}{(2(2\alpha\beta+\alpha-2\beta))}}$ 
 $d^\star_{\text{Phase IVb}} \asymp \mathfrak{f}^{(\alpha-\beta)/(2\alpha\beta+\alpha-2\beta)}$ | | |

Table 5: **SGD-M/SGD: Loss description** $\mathcal{P}(t)$ **for SGD-M** ($\gamma \overset{\text{def}}{=} \gamma_2 + \frac{\gamma_3}{\delta}$) **and SGD** ($\gamma \overset{\text{def}}{=} \gamma_2$) **and compute-optimal curves for** $\tilde{\mathcal{P}}(\frac{\mathfrak{f}}{d \cdot B}, d)$ **across the 4 phases**.

*Proof.* The proof is a straightforward computation. The minimizer of $\hat{\mathcal{P}}(\mathfrak{f}, d)$ in $d$ must occur where the two terms in the maximum are equal, i.e.,

$$\mathcal{C}_0\left(\tfrac{\mathfrak{f}}{d}\right)^{-\gamma_0}d^{-p_0} = \mathcal{C}_1\left(\tfrac{\mathfrak{f}}{d}\right)^{-\gamma_1}d^{-p_1}.$$

Solving for this $d$ gives $d^\star$. Plugging in the value of $d^\star$ into $\hat{\mathcal{P}}(\mathfrak{f}, d)$ gives the optimal value. $\square$

**Remark E.1.** *The possible minimal values of* (72)*, i.e., where pairs of functions in the max are equal, can be reduced further. For instance, if $\mathcal{F}_{ac}(r, d)$ exist for the phase, then for some $0 < r_0 < r_1 < r_2$*

$$\tilde{\mathcal{P}}(t, d) \approx \begin{cases} \mathcal{F}_{pp}(t, d), & 0 < t \leq t_0 \\ \frac{1}{\gamma B}\mathcal{K}_{pp}(t, d) & t_0 < t \leq t_1 \\ \mathcal{F}_{ac}(t, d), & t_1 < t < t_2 \\ \mathcal{F}_0(t, d), & t_2 < t. \end{cases}$$

*Thus, there are only a maximum of three points to check in order to find the optimal compute curve.*

**Remark E.2.** *In view of Lemma E.1, to find the optimal compute curves, we first find the potential curves (i.e., all the possible combinations of two functions in the loss curve are equal while still lying on the loss curve). Then the curve which has the smallest exponent on the flops, $\mathfrak{f}$, is the optimal compute curve.*

## E.1 Stochastic momentum (SGD-M), compute-optimal curves

For constant momentum, the loss curve as well as the forcing terms $\mathcal{F}_0, \mathcal{F}_{pp}, \mathcal{F}_{ac}$ and kernel term $\frac{1}{\gamma_2 B}\mathcal{K}_{pp}$ are entirely similar up to constants as the one for SGD [80]. The compute-optimal curves are hence identical, see Table 5 when replacing $\gamma_2$ by $\gamma_2 + \frac{\gamma_3}{\delta}$ (see Remark G.2). See also Figure 12 for a description of the phases in the $(\alpha, \beta)$-plane.

## E.2 DANA-constant, compute-optimal curves

In all this section, we will use the hyperparameters in Lem. H.5 and Cor. H.2 (see Section 3) with $B = 1$. We discuss the effect of learning rate and batch after. The asymptotics of the forcing and kernel terms for DANA-constant, valid only in some regions of the $(\gamma_2, \gamma_3, B, t, \delta)$ space, are below

$$\mathcal{F}_{pp}(t, d) \asymp \min\{(\gamma_2 Bt)^{-1-\frac{2\beta-1}{2\alpha}}, (\sqrt{\gamma_3 B}t)^{-2-\frac{2\beta-1}{\alpha}}\},$$

$$\mathcal{F}_{ac}(t, d) \asymp d^{-1} \min\{(\gamma_2 Bt)^{-1+\frac{1}{2\alpha}}, (\sqrt{\gamma_3 B}t)^{-2+\frac{1}{\alpha}}\},$$

$$\frac{1}{\gamma_2 B}\mathcal{K}_{pp}(t, d) \asymp \gamma_2 \min\{(\gamma_2 Bt)^{-2+\frac{1}{2\alpha}}, (\sqrt{\gamma_3 B}t)^{-4+\frac{1}{\alpha}}\}, \quad \mathcal{F}_0(t, d) \asymp d^{-2\alpha+\max\{0, 1-2\beta\}}.$$

Derivations for these forcing function and kernel function asymptotics can be found in Section H. For a summary of the compute-optimal curves for DANA-constant, see Table 6 and Figure 12b for a description of the phases in the $(\alpha, \beta)$−plane.

**Below the high-dimensional line, (Phases Ib, Ic, IVa, IVb).** In that case, the limit level $\mathcal{F}_0$ is reached for $t \leq d^{2\alpha} \leq d$. We have, $\forall t \leq d$,

$$\mathcal{F}_{pp}(t, d) \asymp (\gamma_2 Bt)^{-1-\frac{2\beta-1}{2\alpha}}, \quad \mathcal{F}_{ac}(t, d) \asymp d^{-1}(\gamma_2 Bt)^{-1+\frac{1}{2\alpha}},$$

$$\text{and} \quad \frac{1}{\gamma_2 B}\mathcal{K}_{pp}(t, d) \asymp \gamma_2 (\gamma_2 Bt)^{-2+\frac{1}{2\alpha}}.$$

Hence the forcing and kernel terms are similar to SGD and the compute-optimal choices for $d^\star, t^\star$ are the same as in Table 5.

**Above the high-dimensional line (Phases Ia, IIa, IIb, IIIa, IIIb).** A first observation is that the limit on the min in $\mathcal{F}_{pp}, \mathcal{F}_{ac}, \mathcal{K}_{pp}$ can never be compute-optimal by itself, since decreasing $d$ would yield strictly better performance. Hence we only need to check the risk between the different terms of the forcing and kernel. Additionally, from [80], the compute-optimal for SGD happens in all phases for $t \geq d$. Hence it has to be similar for DANA-constant since. We can therefore simplify in the following for $t \geq d$,

$$\mathcal{F}_{pp}(t, d) \asymp (\sqrt{\gamma_3 B}t)^{-2-\frac{2\beta-1}{\alpha}}, \quad \mathcal{F}_{ac}(t, d) \asymp d^{-1}(\sqrt{\gamma_3 B}t)^{-2+\frac{1}{\alpha}},$$

$$\text{and} \quad \frac{1}{\gamma_2 B}\mathcal{K}_{pp}(t, d) \asymp \gamma_2 (\sqrt{\gamma_3 B}t)^{-4+\frac{1}{\alpha}}.$$

**Phase Ia.** In this phase, the approximate loss curve satisfies

$$\mathcal{P}(\tfrac{\mathfrak{f}}{d}, d) \asymp \max\{\mathcal{F}_{pp}(\tfrac{\mathfrak{f}}{d}, d), \mathcal{F}_0(\tfrac{\mathfrak{f}}{d}, d)\} \asymp \max\{(\sqrt{\gamma_3 B}t)^{-2-\frac{2\beta-1}{\alpha}}, d^{-2\alpha-2\beta+1}\}. \tag{73}$$

**Proposition E.1** (Phase I, DANA-constant). *Suppose we are in Phase Ia, i.e. $2\alpha > 1$, $2\beta < 1$. Then the compute-optimal curve $\mathcal{P}(\tfrac{\mathfrak{f}}{d^\star}, d^\star)$ using (73) occurs with $d^\star \asymp \mathfrak{f}^{\frac{1}{3/2+\alpha}}$ and $\mathcal{P}(\tfrac{\mathfrak{f}}{d^\star}, d^\star) \asymp \mathfrak{f}^{-\frac{2\alpha+2\beta-1}{3/2+\alpha}}$.*

*Proof.* We apply Lemma E.1 with

$$\gamma_0 = 2 + \frac{2\beta-1}{\alpha}, \quad p_0 = -1 - \frac{2\beta-1}{2\alpha}, \quad \gamma_1 = 0, \quad \text{and} \quad p_1 = 2\alpha + 2\beta - 1.$$

□

**Phases IIa and IIb.** In this phase, the approximate loss curve satisfies,

$$\mathcal{P}(\tfrac{\mathfrak{f}}{d}, d) \asymp \max\{\mathcal{F}_{pp}(\tfrac{\mathfrak{f}}{d}, d), \mathcal{F}_{ac}(\tfrac{\mathfrak{f}}{d}, d), \mathcal{F}_0(\tfrac{\mathfrak{f}}{d}, d)\}$$

$$\asymp \max\{(\sqrt{\gamma_3 B}t)^{-2-\frac{2\beta-1}{\alpha}}, d^{-1}(\sqrt{\gamma_3 B}t)^{-2+\frac{1}{\alpha}}, d^{-2\alpha+\max\{0, 1-2\beta\}}\}. \tag{74}$$

| | Loss $\mathcal{P}(t)$ | Trade off | Compute-optimal Curves |
|---|---|---|---|
| **Phase I** | $\mathcal{F}_{pp}(t) + \mathcal{F}_0(t)$ | $\mathcal{F}_{pp} = \mathcal{F}_0$ | **Ia** $\tilde{\mathcal{P}}^\star_{\text{Phase Ia}}(\mathfrak{f}) \asymp \mathfrak{f}^{-\frac{2\alpha+2\beta-1}{3/2+\alpha}}$ 
 $d^\star_{\text{Phase Ia}}(\mathfrak{f}) \asymp \mathfrak{f}^{\frac{1}{3/2+\alpha}}$ 

 **Ib** $\tilde{\mathcal{P}}^\star_{\text{Phase Ib}}(\mathfrak{f}) \asymp \mathfrak{f}^{\frac{1}{2}-\alpha-\beta}$ 
 $d^\star_{\text{Phase Ib}}(\mathfrak{f}) \asymp \mathfrak{f}^{\frac{1}{2}}$ 

 **Ic** $\tilde{\mathcal{P}}^\star_{\text{Phase Ic}}(\mathfrak{f}) \asymp \mathfrak{f}^{\frac{\alpha(2\alpha+2\beta-1)}{\alpha(2\beta-3)-2\beta+1}}$ 
 $d^\star_{\text{Phase Ic}}(\mathfrak{f}) \asymp \mathfrak{f}^{\frac{1-2(\alpha+\beta)}{2(\alpha(2\beta-3)-2\beta+1)}}$ |
| **Phase II** | $\mathcal{F}_{pp}(t) + \mathcal{F}_{ac}(t)$ 
 $+\mathcal{F}_0(t)$ | **IIa** $\mathcal{F}_{ac} = \mathcal{F}_0$ 

 **IIb** $\mathcal{F}_{pp} = \mathcal{F}_{ac}$ | $\tilde{\mathcal{P}}^\star_{\text{Phase IIa}}(\mathfrak{f}) \asymp \mathfrak{f}^{-\frac{2\alpha(4\alpha-2)}{4\alpha^2+4\alpha-3}}$ 
 $d^\star_{\text{Phase IIa}}(\mathfrak{f}) \asymp \mathfrak{f}^{\frac{4\alpha-2}{4\alpha^2+4\alpha-3}}$ 

 $\tilde{\mathcal{P}}^\star_{\text{Phase IIb}}(\mathfrak{f}) \asymp \mathfrak{f}^{-\frac{\alpha(2+\frac{2\beta-1}{\alpha})}{3\beta+\alpha}}$ 
 $d^\star_{\text{Phase IIb}}(\mathfrak{f}) \asymp \mathfrak{f}^{\frac{2\beta}{3\beta+\alpha}}$ |
| **Phase III** | $\mathcal{F}_{ac}(t) + \mathcal{F}_0(t)$ 
 $+\dfrac{1}{\gamma_2 B}\mathcal{K}_{pp}(t)$ | **IIIa** $\mathcal{F}_{ac} = \mathcal{F}_0$ 

 **IIIb** $\dfrac{1}{\gamma_2 B}\mathcal{K}_{pp} = \mathcal{F}_{ac}$ | $\tilde{\mathcal{P}}^\star_{\text{Phase IIIa}}(\mathfrak{f}) \asymp \mathfrak{f}^{-\frac{2\alpha(4\alpha-2)}{4\alpha^2+4\alpha-3}}$ 
 $d^\star_{\text{Phase IIIa}}(\mathfrak{f}) \asymp \mathfrak{f}^{\frac{4\alpha-2}{4\alpha^2+4\alpha-3}}$ 

 $\tilde{\mathcal{P}}^\star_{\text{Phase IIIb}}(\mathfrak{f}) \asymp \mathfrak{f}^{-1+\frac{1}{4\alpha}}$ 
 $d^\star_{\text{Phase IIIb}}(\mathfrak{f}) \asymp \mathfrak{f}^{1/2}$ |
| **Phase IV** | $\mathcal{F}_{pp}(t) + \mathcal{F}_0(t)$ 
 $+\dfrac{1}{\gamma_2 B}\mathcal{K}_{pp}(t)$ | **IVa** $\dfrac{1}{\gamma_2 B}\mathcal{K}_{pp} = \mathcal{F}_0$ 

 **IVb** $\dfrac{1}{\gamma_2 B}\mathcal{K}_{pp} = \mathcal{F}_{pp}$ | $\tilde{\mathcal{P}}^\star_{\text{Phase IVa}}(\mathfrak{f}) \asymp \mathfrak{f}^{-\alpha}$ 
 $d^\star_{\text{Phase IVa}}(\mathfrak{f}) \asymp \mathfrak{f}^{1/2}$ 

 $\tilde{\mathcal{P}}^\star_{\text{Phase IVb}}(\mathfrak{f}) \asymp \mathfrak{f}^{\frac{(1-2\alpha)(2\alpha+2\beta-1)}{(2(2\alpha\beta+\alpha-2\beta))}}$ 
 $d^\star_{\text{Phase IVb}} \asymp \mathfrak{f}^{(\alpha-\beta)/(2\alpha\beta+\alpha-2\beta)}$ |

Table 6: **Loss description $\mathcal{P}(t)$ for DANA-constant and compute-optimal curves for $\tilde{\mathcal{P}}(\frac{\mathfrak{f}}{d\cdot B}, d)$ across the 4 phases (and subphases) defined in Figure 12b**. We consider DANA-constant with the hyperparameters in Lem. H.5 and Cor. H.2 (see Section 3) and batch size $B = 1$.

**Proposition E.2** (Phase II, DANA-constant). *Suppose we are in Phase II, i.e. $2\alpha > 1$, $2\beta > 1$, $\alpha > \beta$. Then the compute-optimal curve $\mathcal{P}(\frac{\mathfrak{f}}{d^\star}, d^\star)$ using (74) occurs*

- *if $\alpha > \frac{3}{4}$ (Phase IIa), with $d^\star \asymp \mathfrak{f}^{\frac{1}{3/2+\alpha}}$ and $\mathcal{P}(\frac{\mathfrak{f}}{d^\star}, d^\star) \asymp \mathfrak{f}^{-\frac{2\alpha}{3/2+\alpha}}$,*

- *if $\alpha < \frac{3}{4}$ (Phase IIb), with $d^\star \asymp \mathfrak{f}^{\frac{2\beta}{3\beta+\alpha}}$ and $\mathcal{P}(\frac{\mathfrak{f}}{d^\star}, d^\star) \asymp \mathfrak{f}^{-\frac{(2\alpha+2\beta-1)}{3\beta+\alpha}}$.*

*Proof.* We have two potential cases to check, whether the compute-optimal is attained for $\mathcal{F}_{pp}(t) = \mathcal{F}_{ac}(t)$ or $\mathcal{F}_{ac}(t) = \mathcal{F}_0(t)$.

$\underline{\mathcal{F}_{pp}(t) = \mathcal{F}_{ac}(t)}$: We apply Lemma E.1 with

$$\gamma_0 = 2 + \frac{2\beta - 1}{\alpha}, \quad p_0 = -1 - \frac{2\beta - 1}{2\alpha}, \quad \gamma_1 = 2 - \frac{1}{\alpha}, \quad \text{and} \quad p_1 = \frac{1}{2\alpha}.$$

If $\alpha < \frac{3}{4}$, we have $p_1 - \gamma_1 > 0 > p_0 - \gamma_0$ and it yields an optimal $d^\star_1 = \mathfrak{f}^{\frac{2\beta}{3\beta+\alpha}}$ and $\mathcal{P}(\frac{\mathfrak{f}}{d^\star_1}, d^\star_1) = \mathfrak{f}^{-\frac{(2\alpha+2\beta-1)}{3\beta+\alpha}}$. On the other hand , if $\alpha > \frac{3}{4}$, we have $\mathcal{F}_{pp}(t) = \mathcal{F}_{ac}(t)$ for $t \asymp \mathfrak{f}^{\frac{1/2+\alpha}{3/2+\alpha}}$ but the optimal is to take $t^\star$ the largest. This brings us to the second case.

$\underline{\mathcal{F}_{ac}(t) = \mathcal{F}_0(t)}$: We apply Lemma E.1 with

$$\gamma_0 = 2 - \frac{1}{\alpha}, \quad p_0 = \frac{1}{2\alpha}, \quad \gamma_1 = 0, \quad \text{and} \quad p_1 = 2\alpha.$$

$\square$

If $\alpha > \frac{3}{4}$, we have $p_1 - \gamma_1 > 0 > p_0 - \gamma_0$. Hence the minimum is attained for $d_2^\star = \mathfrak{f}^{\frac{1}{\alpha+3/2}}$ and $\mathcal{P}(\frac{\mathfrak{f}}{d_2^\star}, d_\star^2) = \mathfrak{f}^{-\frac{2\alpha}{\alpha+3/2}}$. For $\alpha < \frac{3}{4}$ however the optimal is to take $t_*$ the smallest. We conclude that for $\alpha > \frac{3}{4}$, in Phase IIa, $d^\star = d_1^\star$ and for $\alpha < \frac{3}{4}$ in Phase IIb, $d^\star = d_2^\star$.

**Phases IIIa and IIIb.** In this phase, the approximate loss curve satisfies,

$$\mathcal{P}(\frac{\mathfrak{f}}{d}, d) \asymp \max\{\frac{1}{\gamma_2 B}\mathcal{K}_{pp}(\frac{\mathfrak{f}}{d}, d), \mathcal{F}_{ac}(\frac{\mathfrak{f}}{d}, d), \mathcal{F}_0(\frac{\mathfrak{f}}{d}, d)\}$$
$$\asymp \max\{\gamma_2(\sqrt{\gamma_3 B}t)^{-4+\frac{1}{\alpha}}, d^{-1}(\sqrt{\gamma_3 B}t)^{-2+\frac{1}{\alpha}}, d^{-2\alpha}\}. \tag{75}$$

**Proposition E.3** (Phase III, DANA-constant). *Suppose we are in Phase III, i.e. $2\alpha > 1$, $2\beta > 1$, $\alpha < \beta$. Then the compute-optimal curve $\mathcal{P}(\frac{\mathfrak{f}}{d^\star}, d^\star)$ using (75) occurs*

- *if $\alpha > \frac{3}{4}$ (Phase IIIa), with $d^\star \asymp \mathfrak{f}^{\frac{4\alpha-2}{4\alpha^2+4\alpha-3}}$ and $\mathcal{P}(\frac{\mathfrak{f}}{d^\star}, d^\star) \asymp \mathfrak{f}^{-\frac{2\alpha(4\alpha-2)}{4\alpha^2+4\alpha-3}}$,*

- *if $\alpha < \frac{3}{4}$ (Phase IIIb), with $d^\star \asymp \mathfrak{f}^{1/2}$ and $\mathcal{P}(\frac{\mathfrak{f}}{d^\star}, d^\star) \asymp \mathfrak{f}^{-1+\frac{1}{4\alpha}}$.*

*Proof.* We have two potential cases to check, whether the compute-optimal is attained for $\frac{1}{\gamma_2 B}\mathcal{K}_{pp}(t, d) = \mathcal{F}_{ac}(t, d)$ or for $\mathcal{F}_{ac}(t, d) = \mathcal{F}_0(t, d)$.

$\underline{\frac{1}{\gamma_2 B}\mathcal{K}_{pp}(t, d) = \mathcal{F}_{ac}(t, d)}$: In that case, we directly know that $d_1^\star : \mathfrak{f}^{1/2}, \mathcal{P}(\frac{\mathfrak{f}}{d^\star}, d^\star) = \mathfrak{f}^{-1+\frac{1}{4\alpha}}$. We also see that by defining

$$\gamma_0 = 4 - \frac{1}{\alpha}, \quad p_0 = -2 + \frac{1}{2\alpha}, \quad \gamma_1 = 2 - \frac{1}{\alpha}, \quad \text{and} \quad p_1 = \frac{1}{2\alpha}.$$

If $\alpha > \frac{3}{4}$, we have $p_0 - \gamma_0 > 0 > p_1 - \gamma_1$ and we apply Lemma E.1 to obtain $d_1^\star : \mathfrak{f}^{1/2}, \mathcal{P}(\frac{\mathfrak{f}}{d_1^\star}, d_1^\star) = \mathfrak{f}^{-1+\frac{1}{4\alpha}}$. If $\alpha < \frac{3}{4}$, the optimal is to choose $t^\star$ the largest which brings us to the other case.

$\underline{\mathcal{F}_{ac}(t, d) = \mathcal{F}_0(t, d)}$: In that case, we define

$$\gamma_0 = 2 - \frac{1}{\alpha}, \quad p_0 = \frac{1}{2\alpha}, \quad \gamma_1 = 0, \quad \text{and} \quad p_1 = 2\alpha.$$

For $\alpha < \frac{3}{4}$, we have $p_1 - \gamma_1 > 0 > p_0 - \gamma_0$ and hence applying Lemma E.1, it brings an optimal $d_2^\star = \mathfrak{f}^{\frac{1}{\alpha+3/2}}$, $\mathcal{P}(\frac{\mathfrak{f}}{d_2^\star}, d_2^\star) = \mathfrak{f}^{-\frac{2\alpha}{\alpha+3/2}}$. For $\alpha > \frac{3}{4}$, the optimal is to choose $t^\star$ the smallest going back to the first case. We conclude that for $\alpha > \frac{3}{4}$, in Phase IIIa, $d^\star = d_2^\star$ and for $\alpha < \frac{3}{4}$ in Phase IIIb, $d^\star = d_1^\star$. $\square$

### E.3 DANA-decaying, compute-optimal curves

In all this section, we will use the hyperparameters in Remark B.3 with $B = 1$. We discuss the effect of learning rate and batch after. We remind below the asymptotics of the forcing and kernel terms, valid only in some regions of the $(\gamma_2, \bar{\gamma}_3, B, t, \delta)$ space ($\tau(t) \asymp (\sqrt{\gamma_3(t)B}t)^2$).

$$\mathcal{F}_{pp}(t, d) \asymp \min\{\gamma_2 Bt, \tau(t)^2\}^{-1-\frac{2\beta-1}{2\alpha}}, \quad \mathcal{F}_{ac}(t, d) \asymp d^{-1}\min\{\gamma_2 Bt, \tau(t)^2\}^{-1+\frac{1}{2\alpha}},$$
$$\frac{1}{\gamma_2 B}\mathcal{K}_{pp}(t, d) \asymp \gamma_2\min\{\gamma_2 Bt, \tau(t)^2\}^{-2+\frac{1}{2\alpha}}, \quad \mathcal{F}_0(t, d) \asymp d^{-2\alpha+\max\{0,1-2\beta\}}.$$

Derivations for these forcing function and kernel function asymptotics can be found in Section I. For a summary of the compute-optimal curves for DANA-constant, see Table 7 and Figure 12d for a description of the phases in the $(\alpha, \beta)$−plane.

| | Loss $\mathcal{P}(t)$ | Trade off | | Compute-optimal Curves |
|---|---|---|---|---|
| **Phase I** | $\mathcal{F}_{pp}(t) + \mathcal{F}_0(t)$ | $\mathcal{F}_{pp} = \mathcal{F}_0$ | **Ia** | $\tilde{\mathcal{P}}^\star_{\text{Phase Ia}}(\mathfrak{f}) \asymp \mathfrak{f}^{\frac{(1-2\alpha-2\beta)(4\alpha-1)}{(4\alpha-1)+4\alpha^2}}$ $d^\star_{\text{Phase Ia}}(\mathfrak{f}) \asymp \mathfrak{f}^{\frac{4\alpha-1}{4\alpha^2+4\alpha-1}}$ |
| | | | **Ib** | $\tilde{\mathcal{P}}^\star_{\text{Phase Ib}}(\mathfrak{f}) \asymp \mathfrak{f}^{\frac{1}{2}-\alpha-\beta}$ $d^\star_{\text{Phase Ib}}(\mathfrak{f}) \asymp \mathfrak{f}^{\frac{1}{2}}$ |
| | | | **Ic** | $\tilde{\mathcal{P}}^\star_{\text{Phase Ic}}(\mathfrak{f}) \asymp \mathfrak{f}^{\frac{\alpha(2\alpha+2\beta-1)}{\alpha(2\beta-3)-2\beta+1}}$ $d^\star_{\text{Phase Ic}}(\mathfrak{f}) \asymp \mathfrak{f}^{\frac{1-2(\alpha+\beta)}{2(\alpha(2\beta-3)-2\beta+1)}}$ |
| **Phase II** | $\mathcal{F}_{pp}(t) + \mathcal{F}_{ac}(t) + \mathcal{F}_0(t)$ | **IIa** $\;\mathcal{F}_{ac} = \mathcal{F}_0$ | | $\tilde{\mathcal{P}}^\star_{\text{Phase IIa}}(\mathfrak{f}) \asymp \mathfrak{f}^{-\frac{2\alpha(4\alpha-1)}{4\alpha-1+4\alpha^2}}$ $d^\star_{\text{Phase IIa}}(\mathfrak{f}) \asymp \mathfrak{f}^{\frac{4\alpha-1}{4\alpha-1+4\alpha^2}}$ |
| | | **IIb** $\;\mathcal{F}_{pp} = \mathcal{F}_{ac}$ | | $\tilde{\mathcal{P}}^\star_{\text{Phase IIb}}(\mathfrak{f}) \asymp \mathfrak{f}^{-\frac{(2\alpha+2\beta-1)(4\alpha-1)}{2(2\alpha^2+4\alpha\beta-\beta)}}$ $d^\star_{\text{Phase IIb}}(\mathfrak{f}) \asymp \mathfrak{f}^{\frac{(4\alpha-1)\beta}{2\alpha^2+4\alpha\beta-\beta}}$ |
| **Phase III** | $\mathcal{F}_{ac}(t) + \mathcal{F}_0(t) + \frac{1}{\gamma_2 B}\mathcal{K}_{pp}(t)$ | **IIIa** $\;\mathcal{F}_{ac} = \mathcal{F}_0$ | | $\tilde{\mathcal{P}}^\star_{\text{Phase IIIa}}(\mathfrak{f}) \asymp \mathfrak{f}^{-\frac{2\alpha(4\alpha-1)}{4\alpha-1+4\alpha^2}}$ $d^\star_{\text{Phase IIIa}}(\mathfrak{f}) \asymp \mathfrak{f}^{\frac{4\alpha-1}{4\alpha-1+4\alpha^2}}$ |
| | | **IIIb** $\;\frac{1}{\gamma_2 B}\mathcal{K}_{pp} = \mathcal{F}_{ac}$ | | $\tilde{\mathcal{P}}^\star_{\text{Phase IIIb}}(\mathfrak{f}) \asymp \mathfrak{f}^{-\frac{(4\alpha-1)^2}{2\alpha(6\alpha-1)}}$ $d^\star_{\text{Phase IIIb}}(\mathfrak{f}) \asymp \mathfrak{f}^{\frac{4\alpha-1}{6\alpha-1}}$ |
| **Phase IV** | $\mathcal{F}_{pp}(t) + \mathcal{F}_0(t) + \frac{1}{\gamma_2 B}\mathcal{K}_{pp}(t)$ | **IVa** $\;\frac{1}{\gamma_2 B}\mathcal{K}_{pp} = \mathcal{F}_0$ | | $\tilde{\mathcal{P}}^\star_{\text{Phase IVa}}(\mathfrak{f}) \asymp \mathfrak{f}^{-\alpha}$ $d^\star_{\text{Phase IVa}}(\mathfrak{f}) \asymp \mathfrak{f}^{1/2}$ |
| | | **IVb** $\;\frac{1}{\gamma_2 B}\mathcal{K}_{pp} = \mathcal{F}_{pp}$ | | $\tilde{\mathcal{P}}^\star_{\text{Phase IVb}}(\mathfrak{f}) \asymp \mathfrak{f}^{\frac{(1-2\alpha)(2\alpha+2\beta-1)}{(2(2\alpha\beta+\alpha-2\beta))}}$ $d^\star_{\text{Phase IVb}} \asymp \mathfrak{f}^{(\alpha-\beta)/(2\alpha\beta+\alpha-2\beta)}$ |

Table 7: **DANA-decaying: Loss description for $\mathcal{P}(t)$ when solved with the DANA-decaying algorithm and compute-optimal curves for $\tilde{\mathcal{P}}(\frac{\mathfrak{f}}{d \cdot B}, d)$ across the 4 phases (and sub-phases) defined in Figure 12d**. We consider DANA-decaying with the hyperparameters in Remark B.3 and batch size $B = 1$.

**Below the high-dimensional line, (Phases Ib, Ic, IVa, IVb).** In that case, $\gamma_2 Bt \lesssim \bar{\gamma}_3 Bt^{2-1/(2\alpha)} \lesssim \tau(t)^2$. Hence, we can write

$$\mathcal{F}_{pp}(t,d) \asymp (\gamma_2 Bt)^{-1-\frac{2\beta-1}{2\alpha}}, \quad \mathcal{F}_{ac}(t,d) \asymp d^{-1}(\gamma_2 Bt)^{-1+\frac{1}{2\alpha}},$$

$$\text{and} \quad \frac{1}{\gamma_2 B}\mathcal{K}_{pp}(t,d) \asymp \gamma_2(\gamma_2 Bt)^{-2+\frac{1}{2\alpha}}.$$

Hence the forcing and kernel terms are similar to SGD and the compute-optimal choices for $d^\star, t^\star$ are the same as in Table 5.

**Above the high-dimensional line (Phases Ia, IIa, IIb, IIIa, IIIb).** In that case, we can check that $\gamma_2 Bt \gtrsim \bar{\gamma}_3 Bt^{2-1/(2\alpha)} \gtrsim \tau(t)^2$. Therefore we simplify

$$\mathcal{F}_{pp}(t,d) \asymp (\bar{\gamma}_3 Bt^{2-1/(2\alpha)})^{-1-\frac{2\beta-1}{2\alpha}}, \quad \mathcal{F}_{ac}(t,d) \asymp d^{-1}(\bar{\gamma}_3 Bt^{2-1/(2\alpha)})^{-1+\frac{1}{2\alpha}},$$

$$\text{and} \quad \frac{1}{\gamma_2 B}\mathcal{K}_{pp}(t,d) \asymp \gamma_2(\bar{\gamma}_3 Bt^{2-1/(2\alpha)})^{-2+\frac{1}{2\alpha}}.$$

**Phase Ia.** In this phase, the approximate loss curve satisfies

$$\mathcal{P}(\tfrac{\mathfrak{f}}{d}, d) \asymp \max\{\mathcal{F}_{pp}(\tfrac{\mathfrak{f}}{d}, d), \mathcal{F}_0(\tfrac{\mathfrak{f}}{d}, d)\} \asymp \max\{(\bar{\gamma}_3 Bt^{2-1/(2\alpha)})^{-1-\frac{2\beta-1}{2\alpha}}, d^{-2\alpha-2\beta+1}\}. \quad (76)$$

**Proposition E.4** (Phase Ia, DANA-decaying). *Suppose we are in Phase Ia, i.e. $2\alpha > 1$, $2\beta < 1$. Then the compute-optimal curve $\mathcal{P}(\frac{\mathfrak{f}}{d^\star}, d^\star)$ using (76) occurs with $d^\star \asymp \mathfrak{f}^{\frac{4\alpha-1}{4\alpha^2+4\alpha-1}}$ and $\mathcal{P}(\frac{\mathfrak{f}}{d^\star}, d^\star) \asymp \mathfrak{f}^{\frac{(1-2\alpha-2\beta)(4\alpha-1)}{(4\alpha-1)+4\alpha^2}}$.*

*Proof.* We apply Lemma E.1 with

$$\gamma_0 = \left(2 - \frac{1}{2\alpha}\right)\left(1 + \frac{2\beta-1}{2\alpha}\right), \quad p_0 = 0, \gamma_1 = 0, \quad \text{and} \quad p_1 = 2\alpha + 2\beta - 1.$$

$\square$

**Phases IIa and IIb.** In this case, the approximate loss curve satisfies

$$
\begin{aligned}
\mathcal{P}(\tfrac{\mathfrak{f}}{d}, d) &\asymp \max\{\mathcal{F}_{pp}(\tfrac{\mathfrak{f}}{d}, d), \mathcal{F}_{ac}(\tfrac{\mathfrak{f}}{d}, d), \mathcal{F}_0(\tfrac{\mathfrak{f}}{d}, d)\} \\
&\asymp \max\{(\bar{\gamma}_3 B t^{2-1/(2\alpha)})^{-1-\frac{2\beta-1}{2\alpha}}, d^{-1}(\bar{\gamma}_3 B t^{2-1/(2\alpha)})^{-1+\frac{1}{2\alpha}}, d^{-2\alpha}\}.
\end{aligned}
\tag{77}
$$

**Proposition E.5** (Phase II, DANA-decaying). *Suppose we are in Phase II, i.e. $2\alpha > 1$, $2\beta > 1$, $\alpha > \beta$. Then the compute-optimal curve $\mathcal{P}(\frac{\mathfrak{f}}{d^\star}, d^\star)$ using (77) occurs*

- *if $\alpha > \frac{3+\sqrt{5}}{4}$ (Phase IIa), with $d^\star \asymp \mathfrak{f}^{\frac{4\alpha-1}{4\alpha-1+4\alpha^2}}$ and $\mathcal{P}(\frac{\mathfrak{f}}{d^\star}, d^\star) \asymp \mathfrak{f}^{-\frac{2\alpha(4\alpha-1)}{4\alpha-1+4\alpha^2}}$,*

- *if $\alpha < \frac{3+\sqrt{5}}{4}$ (Phase IIb), with $d^\star \asymp \mathfrak{f}^{\frac{(4\alpha-1)\beta}{2\alpha^2+4\alpha\beta-\beta}}$ and $\mathcal{P}(\frac{\mathfrak{f}}{d^\star}, d^\star) \asymp \mathfrak{f}^{-\frac{(2\alpha+2\beta-1)(4\alpha-1)}{2(2\alpha^2+4\alpha\beta-\beta)}}$.*

*Proof.* The compute-optimal choice can be either attained for $\mathcal{F}_{pp}(\frac{\mathfrak{f}}{d}, d) = \mathcal{F}_{ac}(\frac{\mathfrak{f}}{d}, d)$ or $\mathcal{F}_{ac}(\frac{\mathfrak{f}}{d}, d) = \mathcal{F}_0(\frac{\mathfrak{f}}{d}, d)$.

$\underline{\mathcal{F}_{pp}(\frac{\mathfrak{f}}{d}, d) = \mathcal{F}_{ac}(\frac{\mathfrak{f}}{d}, d)}$: In that case, we introduce

$$\gamma_0 = \left(2 - \frac{1}{2\alpha}\right)\left(1 + \frac{2\beta-1}{2\alpha}\right), \quad p_0 = 0, \quad \gamma_1 = \left(2 - \frac{1}{2\alpha}\right)\left(1 - \frac{1}{2\alpha}\right), \quad \text{and} \quad p_1 = 1.$$

If $\alpha < \frac{3+\sqrt{5}}{4}$ then $p_1 - \gamma_1 > 0 > p_0 - \gamma_0$ and applying Lemma E.1 we obtain an optimal $d_1^\star \asymp \mathfrak{f}^{\frac{(4\alpha-1)\beta}{2\alpha^2+4\alpha\beta-\beta}}$ and $\mathcal{P}^\star(\frac{\mathfrak{f}}{d_1^\star}, d_1^\star) \asymp \mathfrak{f}^{-\frac{(2\alpha+2\beta-1)(4\alpha-1)}{2(2\alpha^2+4\alpha\beta-\beta)}}$. However, if $\alpha > \frac{3+\sqrt{5}}{4}$ then $p_1 - \gamma_1 < 0$ and $p_0 - \gamma_0 < 0$. Hence the optimal is to choose $t^\star$ the largest which brings us to the second case.

$\underline{\mathcal{F}_{ac}(\frac{\mathfrak{f}}{d}, d) = \mathcal{F}_0(\frac{\mathfrak{f}}{d}, d)}$: In that case we define

$$\gamma_0 = \left(2 - \frac{1}{2\alpha}\right)\left(1 - \frac{1}{2\alpha}\right), \quad p_0 = 1, \quad \gamma_1 = 0, \quad \text{and} \quad p_1 = 2\alpha.$$

If $\alpha > \frac{3+\sqrt{5}}{4}$, then $p_0 - \gamma_0 < 0 < p_1 - \gamma_1$ and we apply Lemma E.1 to obtain that $d_2^\star \asymp \mathfrak{f}^{\frac{4\alpha-1}{4\alpha-1+4\alpha^2}}$ and $\mathcal{P}(\frac{\mathfrak{f}}{d_2^\star}, d_2^\star) \asymp \mathfrak{f}^{-\frac{2\alpha(4\alpha-1)}{4\alpha-1+4\alpha^2}}$. On the other hand, if $\alpha < \frac{3+\sqrt{5}}{4}$, then $p_0 - \gamma_0 > 0$, $p_1 - \gamma_1 > 0$ and the compute optimal is to take $t^\star$ the smallest, i.e. going back to the first case. We conclude that for $\alpha > \frac{3+\sqrt{5}}{4}$, in Phase IIIa, $d^\star = d_1^\star$ and for $\frac{3+\sqrt{5}}{4}$ in Phase IIIb, $d^\star = d_2^\star$. $\square$

**Phases IIIa and IIIb.** In this case, the approximate loss curve satisfies

$$
\begin{aligned}
\mathcal{P}(\tfrac{\mathfrak{f}}{d}, d) &\asymp \max\{\tfrac{1}{\gamma_2 B}\mathcal{K}_{pp}(\tfrac{\mathfrak{f}}{d}, d), \mathcal{F}_{ac}(\tfrac{\mathfrak{f}}{d}, d), \mathcal{F}_0(\tfrac{\mathfrak{f}}{d}, d)\} \\
&\asymp \max\{\bar{\gamma}_3(\bar{\gamma}_3 B t^{2-1/(2\alpha)})^{-2+\frac{1}{2\alpha}}, d^{-1}(\bar{\gamma}_3 B t^{2-1/(2\alpha)})^{-1+\frac{1}{2\alpha}}, d^{-2\alpha}\}.
\end{aligned}
\tag{78}
$$

**Proposition E.6** (Phase III, DANA-decaying). *Suppose we are in Phase III, i.e. $2\alpha > 1$, $2\beta > 1$, $\alpha < \beta$. Then the compute-optimal curve $\mathcal{P}(\frac{\mathfrak{f}}{d^\star}, d^\star)$ using (78) occurs*

- *if $\alpha > \frac{3+\sqrt{5}}{4}$ (Phase IIIa), with $d^\star \asymp \mathfrak{f}^{\frac{4\alpha-1}{4\alpha-1+4\alpha^2}}$ and $\mathcal{P}(\frac{\mathfrak{f}}{d^\star}, d^\star) \asymp \mathfrak{f}^{-\frac{2\alpha(4\alpha-1)}{4\alpha-1+4\alpha^2}}$,*

- *if $\alpha < \frac{3+\sqrt{5}}{4}$ (Phase IIIb), with $d^\star \asymp \mathfrak{f}^{\frac{4\alpha-1}{6\alpha-1}}$ and $\mathcal{P}(\frac{\mathfrak{f}}{d^\star}, d^\star) \asymp \mathfrak{f}^{-\frac{(4\alpha-1)^2}{2\alpha(6\alpha-1)}}$.*

*Proof.* The compute-optimal choice can be either attained for $\mathcal{F}_{pp}(\frac{\mathfrak{f}}{d}, d) = \mathcal{F}_{ac}(\frac{\mathfrak{f}}{d}, d)$ or $\mathcal{F}_{ac}(\frac{\mathfrak{f}}{d}, d) = \mathcal{F}_0(\frac{\mathfrak{f}}{d}, d)$.

$\underline{\frac{1}{\gamma_2 B}\mathcal{K}_{pp}(\frac{\mathfrak{f}}{d}, d) = \mathcal{F}_{ac}(\frac{\mathfrak{f}}{d}, d)}$: In that case, we introduce

$$\gamma_0 = \left(2 - \frac{1}{2\alpha}\right)\left(2 - \frac{1}{2\alpha}\right), \quad p_0 = 0, \quad \gamma_1 = \left(2 - \frac{1}{2\alpha}\right)\left(1 - \frac{1}{2\alpha}\right), \quad \text{and} \quad p_1 = 1.$$

If $\alpha < \frac{3+\sqrt{5}}{4}$, then $p_1 - \gamma_1 > 0 > p_0 - \gamma_0$ and we apply Lemma E.1 to obtain $d_*^1 = \mathfrak{f}^{\frac{4\alpha-1}{6\alpha-1}}$ and $\mathcal{P}(\frac{\mathfrak{f}}{d_1^\star}, d_1^\star) \asymp \mathfrak{f}^{-\frac{(4\alpha-1)^2}{2\alpha(6\alpha-1)}}$. However, when $\alpha > \frac{3+\sqrt{5}}{4}$, then $p_0 - \gamma_0 < 0$, $p_1 - \gamma_1 < 0$ and the optimal choice is to take $t^\star$ the largest, leading to the second case.

$\underline{\mathcal{F}_{ac}(\frac{\mathfrak{f}}{d}, d) = \mathcal{F}_0(\frac{\mathfrak{f}}{d}, d)}$: In that case we define

$$\gamma_0 = \left(2 - \frac{1}{2\alpha}\right)\left(1 - \frac{1}{2\alpha}\right), \quad p_0 = 1, \quad \gamma_1 = 0, \quad \text{and} \quad p_1 = 2\alpha.$$

If $\alpha > \frac{3+\sqrt{5}}{4}$, then $p_0 - \gamma_0 < 0 < p_1 - \gamma_1$ and we apply Lemma E.1 to obtain that $d_2^\star \asymp \mathfrak{f}^{\frac{4\alpha-1}{4\alpha-1+4\alpha^2}}$ and $\mathcal{P}(\frac{\mathfrak{f}}{d_2^\star}, d_2^\star) \asymp \mathfrak{f}^{-\frac{2\alpha(4\alpha-1)}{2(2\alpha^2+2\alpha-1/2)}}$. On the other hand, if $\alpha < \frac{3+\sqrt{5}}{4}$, then $p_0 - \gamma_0 > 0$, $p_1 - \gamma_1 > 0$ and the compute optimal is to take $t^\star$ the smallest, i.e. going back to the first case. We conclude that for $\alpha > \frac{3+\sqrt{5}}{4}$, in phase IIIa, $d^\star = d_1^\star$ and for $\frac{3+\sqrt{5}}{4}$ in Phase IIIb, $d^\star = d_2^\star$. $\qquad \square$

## E.4 Comparison of samples needed at compute optimality

Independently of compute, a bottleneck in the training of large models is the amount of data available. While the size of a model can be arbitrarily increased, data comes with hard limits: the size of internet when it comes to language models [97] or when dealing with resource constrained tasks such as medical imaging [13]. Hence a natural question is *how do the previous algorithms compare in terms of samples used at compute-optimality?* In what follows, we denote DANA-c as DANA-constant and DANA-d as DANA-decaying.

We know that for a given number of samples/iterations $t \geq 0$, DANA-d always achieve smaller or equal loss than DANA-c which in turn achieves smaller or equal loss than SGD. Hence for a fixed given loss level, DANA-decay needs less samples to achieve this loss than DANA-constant which needs less than SGD; strictly less when the scaling laws exponents are improved in the considered regime. However, it is not clear that this ordering remains the same when considering the compute-optimal training regime. Indeed, in some regions of the $(\alpha, \beta)$ plane, the corresponding training regimes can be different for two algorithms. For example, for $\frac{3}{4} < \alpha$ and $\beta < \alpha$ the compute-optimal regime of DANA-c happens at the frontier $\mathcal{F}_{ac}/\mathcal{F}_0$ while the compute-optimal regime for SGD is between $\mathcal{K}_{pp}/\mathcal{F}_{ac}$. The compute-optimal regime of DANA-c is shifted later in training (this is a general effect of the acceleration, see Remark E.2). Hence, even if for a given loss level DANA-c needs less samples than SGD, DANA-c needs more samples at compute-optimality.

In the following denote for $i \in \{\text{SGD}, \text{DANA-c}, \text{DANA-d}\}$, $\rho_i > 0$ such that in a given phase, the number of samples required for algorithm $i$ is at compute optimality $t^\star = \mathfrak{f}^{\rho_i}$. The smallest $\rho_i$, the more *data-efficient* the algorithm is for a given compute. A first observation, is that even though the loss $\mathcal{P}^\star(\mathfrak{f})$ is continuous, across all the phases, neither the optimal dimension $d^\star(\mathfrak{f})$ nor the number of

samples $t^\star(\mathfrak{f})$ are continuous. The points of discontinuity are exactly when the trade-off condition changes, for examples at the borders $IIa/IIb$ or $IIIa/IIIb$.

***Claim*** DANA-d will always use less samples at compute-optimality than DANA-c in all phases. Additionally, given $(\alpha, \beta)$ and for two algorithms in {SGD, DANA-c, DANA-d}, **if compute-optimality is reached at the same trade-off** between $\mathcal{F}_0, \mathcal{F}_{ac}, \mathcal{F}_{pp}, \frac{1}{\gamma_2 B}\mathcal{K}_{pp}$ then DANA-decaying uses less samples than DANA-constant which in turn uses less samples than SGD at compute-optimality. The improvement is strict if one of the algorithm is accelerating with respect to the other. Hence DANA-d, DANA-c and SGD are in that order sample efficient in phases Ia, IIb , IIIb. However, for large $\alpha$, compute-optimality for DANA-c, DANA-d is reached later in training due to the acceleration, and DANA-c, DANA-d may as well use more than less samples than SGD at compute-optimality, depending on $\alpha, \beta$.

*Proof of the claim.* We consider each phase:

- Below the high-dimensional line, SGD, DANA-constant and DANA-decaying have the same scaling laws.

- In Phase Ia, $\rho_{\text{SGD}} = 1 - \frac{1}{2\alpha+1} > \rho_{\text{DANA-c}} = 1 - \frac{1}{3/2+\alpha} > \rho_{\text{DANA-d}} = 1 - \frac{4\alpha-1}{4\alpha^2+4\alpha-1}$.

- In Phase IIb (of DANA-c), $\rho_{\text{SGD}} = 1 - \frac{\beta/\alpha}{1+\beta/\alpha} > \rho_{\text{DANA-c}} = 1 - \frac{2\beta}{3\beta+\alpha} > \rho_{\text{DANA-d}} = 1 - \frac{(4\alpha-1)\beta}{2\alpha^2+4\alpha\beta-\beta}$.

- In Phase IIIb (of DANA-c), $\rho_{\text{SGD}} = \frac{1}{2} = \rho_{\text{DANA-c}} > \rho_{\text{DANA-d}} = 1 - \frac{4\alpha-1}{6\alpha-1}$.

- In Phase IIb (of DANA-d), $\rho_{\text{SGD}} = 1 - \frac{\beta/\alpha}{1+\beta/\alpha} > \rho_{\text{DANA-d}} = 1 - \frac{(4\alpha-1)\beta}{2\alpha^2+4\alpha\beta-\beta}$. I we are additionally in phase IIa, then $\rho_{\text{DANA-c}} = 1 - \frac{4\alpha-2}{4\alpha^2+4\alpha-3} > \rho_{\text{DANA-d}} = 1 - \frac{(4\alpha-1)\beta}{2\alpha^2+4\alpha\beta-\beta}$.

- In Phase IIIb (of DANA-d), $\rho_{\text{SGD}} = \frac{1}{2} > \rho_{\text{DANA-d}} = 1 - \frac{4\alpha-1}{6\alpha-1}$. If we are additionally in Phase IIIa, then $\rho_{\text{DANA-c}} = 1 - \frac{4\alpha-2}{4\alpha^2+4\alpha-3} > \rho_{\text{DANA-d}} = 1 - \frac{4\alpha-1}{6\alpha-1}$.

- In Phase IIa (of DANA-d), $\rho_{\text{DANA-c}} = 1 - \frac{4\alpha-2}{4\alpha^2+4\alpha-3} > \rho_{\text{DANA-d}} = 1 - \frac{4\alpha-1}{4\alpha^2+4\alpha-1}$.

- In Phase IIIa (of DANA-d), $\rho_{\text{DANA-c}} = 1 - \frac{4\alpha-2}{4\alpha^2+4\alpha-3} > \rho_{\text{DANA-d}} = 1 - \frac{4\alpha-1}{4\alpha^2+4\alpha-1}$.

- In the other cases, there is no general rule of whether one algorithm uses less samples than the other at compute-optimality. The interested reader may still easily derive them using Tables 5 to 7.

One can explain why *DANA-d always use less samples than DANA-c*, even when they don't share the same trade-off condition (i.e. for $\alpha \in \left[\frac{3}{4}, \frac{3+\sqrt{5}}{4}\right]$, $\beta > \frac{1}{2}$) by the fact that DANA-c shifts the trade-off (later in training) **for smaller $\alpha$ than DANA-d**. $\qquad\square$

### E.5 Summary on compute-optimality results

We provide some specific details about compute-optimality for each algorithm in the different phases.

*Below high-dimensional line (Phases Ib, Ic, IVa, IVb):* Limit level $\mathcal{F}_0$ is reached when $t \leq d$. DANA/SGD-M has same scaling laws as SGD at compute-optimality.

**SGD-M.** Since same scaling law as SGD, there is a large portion of the $(\alpha, \beta)$-plane (Phase III, IVa, Ib) where we have universal scaling, i.e., *params exponent* is $\mathfrak{f}^{1/2}$ or equivalently, the compute-optimal regime is the same as the proportional regime ($t \asymp d$). This was observed empirically in [50].

**DANA-constant/DANA-decaying.** To improve the compute-optimal loss exponent for DANA-constant, compute-optimality must occur after iteration $d^{-1}$ (Thm. H.3). As noted in Thm. I.2, DANA-decaying improves the loss exponents for *all* scaling regimes where $2\alpha > 1$, including the compute-optimality regime.

***Phase Ia:*** Here, $\mathcal{F}_{pp}$ accelerates beginning at $t \geq d$ until it reaches the limit risk $\mathcal{F}_0$. While the compute-optimal tradeoff occurs at $\mathcal{F}_{pp}$ and $\mathcal{F}_0$ (same as SGD), DANA-constant reaches this point faster than SGD; thus DANA-constant outscales SGD.

***Phase II:*** This phase involves $\mathcal{F}_{pp}$, $\mathcal{F}_{ac}$, $\mathcal{F}_0$ and at compute-optimality, we always see a better loss exponent since the tradeoff point occurs at a point where $t \gtrsim d$. Notably the acceleration changes the tradeoff constraints (e.g., where compute-optimal tradeoff occurs). For $\alpha \leq 0.75$ (Phase IIb, DANA-constant) or $\alpha \leq (3 + \sqrt{5})/4$ (Phase IIb, DANA-decaying), tradeoff occurs at the same two terms as SGD, i.e., $\mathcal{F}_{pp}$ and $\mathcal{F}_{ac}$, but its get there faster. This means that one uses *fewer* samples than SGD to achieve compute-optimality. For $\alpha \geq 0.75$ (Phase IIa, DANA-constant) or $\alpha \geq (3 + \sqrt{5})/4$ (Phase IIa, DANA-decaying), the acceleration of $\mathcal{F}_{ac}$ shifts the compute-optimal frontier to $\mathcal{F}_{ac}$ and $\mathcal{F}_0$, making DANA model capacity constrained, that is, why the compute-optimal frontier exists changes.

***Phase III:*** This phase involves $\mathcal{K}_{pp}$, $\mathcal{F}_{ac}$, and $\mathcal{F}_0$ and, notably, $\mathcal{K}_{pp}$ and $\mathcal{F}_{ac}$ always intersect at $t \asymp d$. The same as Phase II occurs with $\mathcal{K}_{pp}$ replacing $\mathcal{F}_{pp}$. When $\alpha < \frac{3}{4}$, though, for DANA-constant the compute-optimal tradeoff occurs at $\mathcal{K}_{pp}$ and $\mathcal{F}_{ac}$, so no outscaling SGD.

We summarize the results of Section E below; for Phase Diagrams see Figure 12, for scaling laws and compute-optimal tradeoffs see Figure 13 and 14 (above the high-dimensional line) and Figure 15 (below the high-dimensional line).

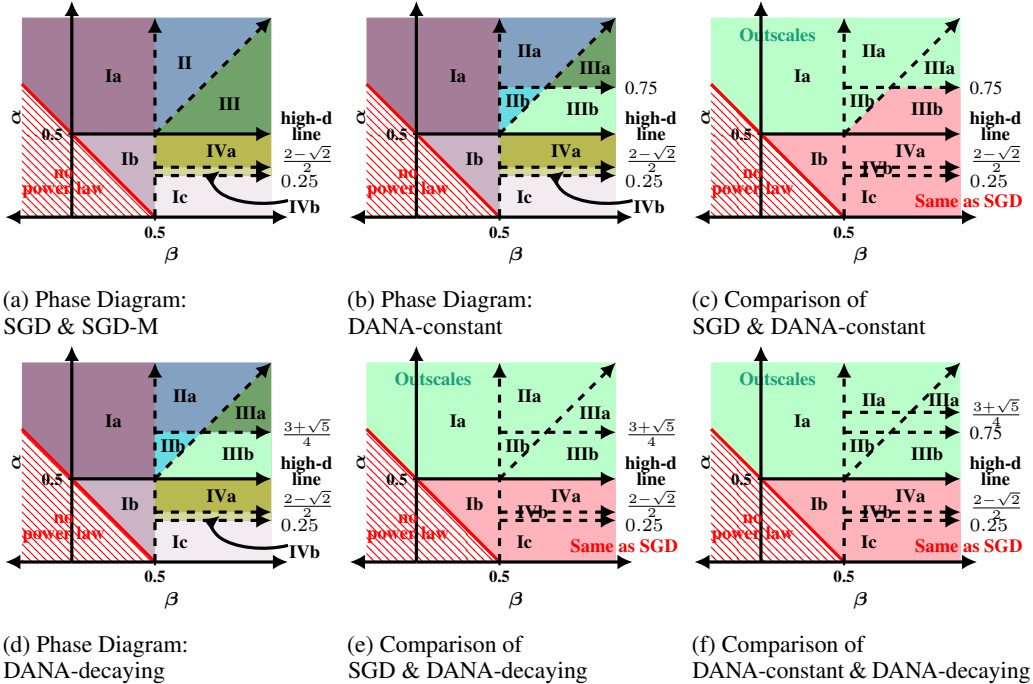

Figure 12: **Phase diagrams for the various momentum algorithms in the compute-optimal regime.** See Figure 13, Figure 14, and Figure 15 for cartoon pictures of the scaling laws and tradeoffs across all phases and algorithms. The main phases (I,II,III,IVa) are based on the components of the loss that dominate at each time. The phases are further broken down to account for the different tradeoffs in the compute-optimal training regime. We always use $\kappa_1 = \max\{0, 1 - 2\alpha\}$ and we use DANA-constant with $\kappa_2 = 1 + \kappa_1$ and $\kappa_3 = 0$ and DANA-decaying with $\kappa_3 = 1/(2\alpha)$, $\kappa_2 = \kappa_1$, and batch size $B = 1$.

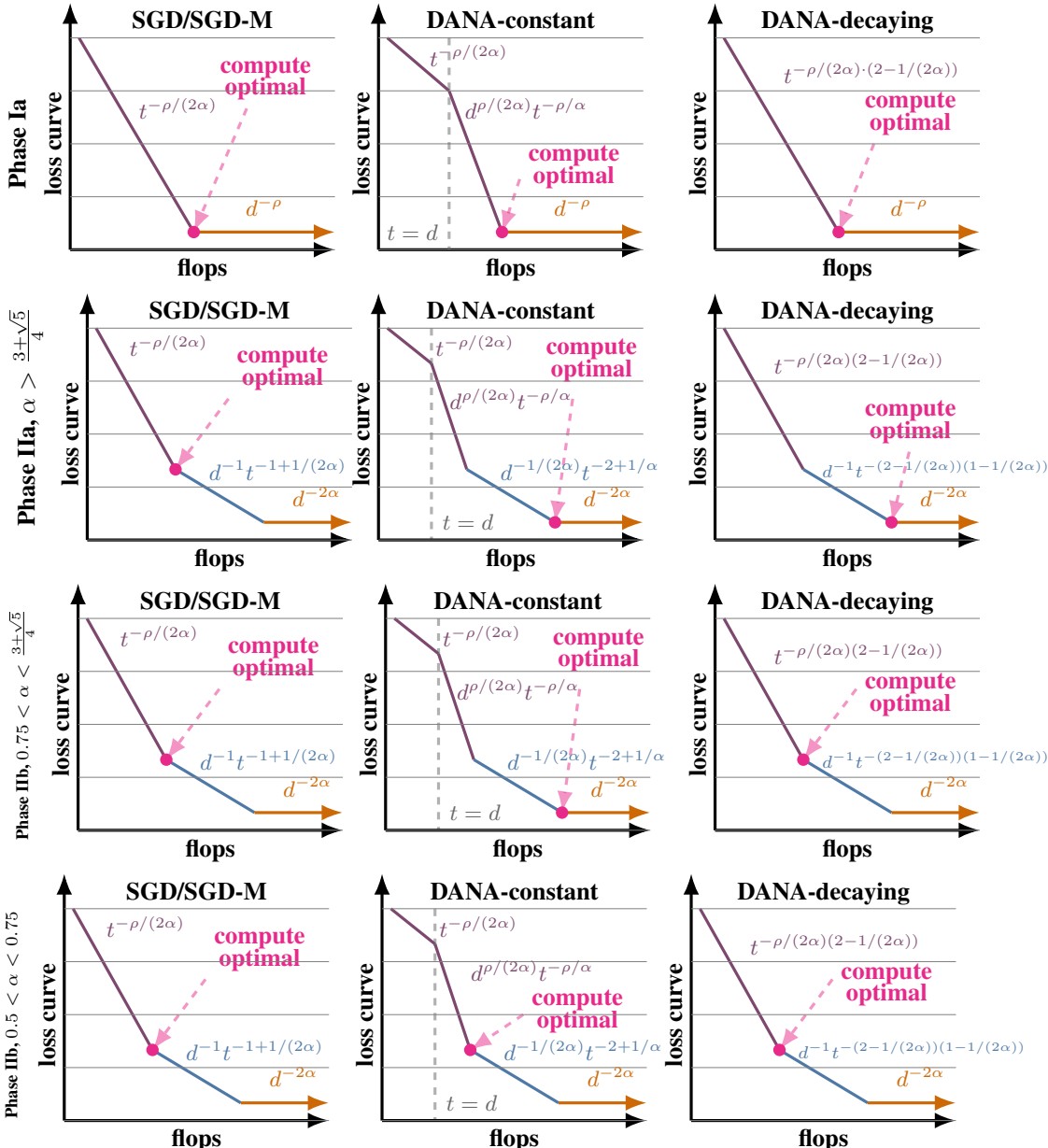

Figure 13: **Cartoon plots of the scaling laws for each algorithm above the high-dimensional line (Phase Ia, IIa, IIb).** Pictures for the scaling laws for all three algorithms in each of the different phases. When $t < d$, DANA-constant behaves like SGD/SGD-M. Observe that the trade off point for compute-optimum changes across phases and algorithms; indicated by (magenta). population bias, $\mathcal{F}_{pp}(t) =$ (purple), embedding bias, $\mathcal{F}_{ac}(t) =$ (blue), and model capacity, $\mathcal{F}_0(t) =$ (orange). Variance due to the algorithm has no impact. **Here** $\rho = 2\alpha + 2\beta - 1$. We always use $\kappa_1 = \max\{0, 1 - 2\alpha\}$ and we use DANA-constant with $\kappa_2 = 1 + \kappa_1$ and $\kappa_3 = 0$ and DANA-decaying with $\kappa_3 = 1/(2\alpha)$, $\kappa_2 = \kappa_1$, and batch size $B = 1$.

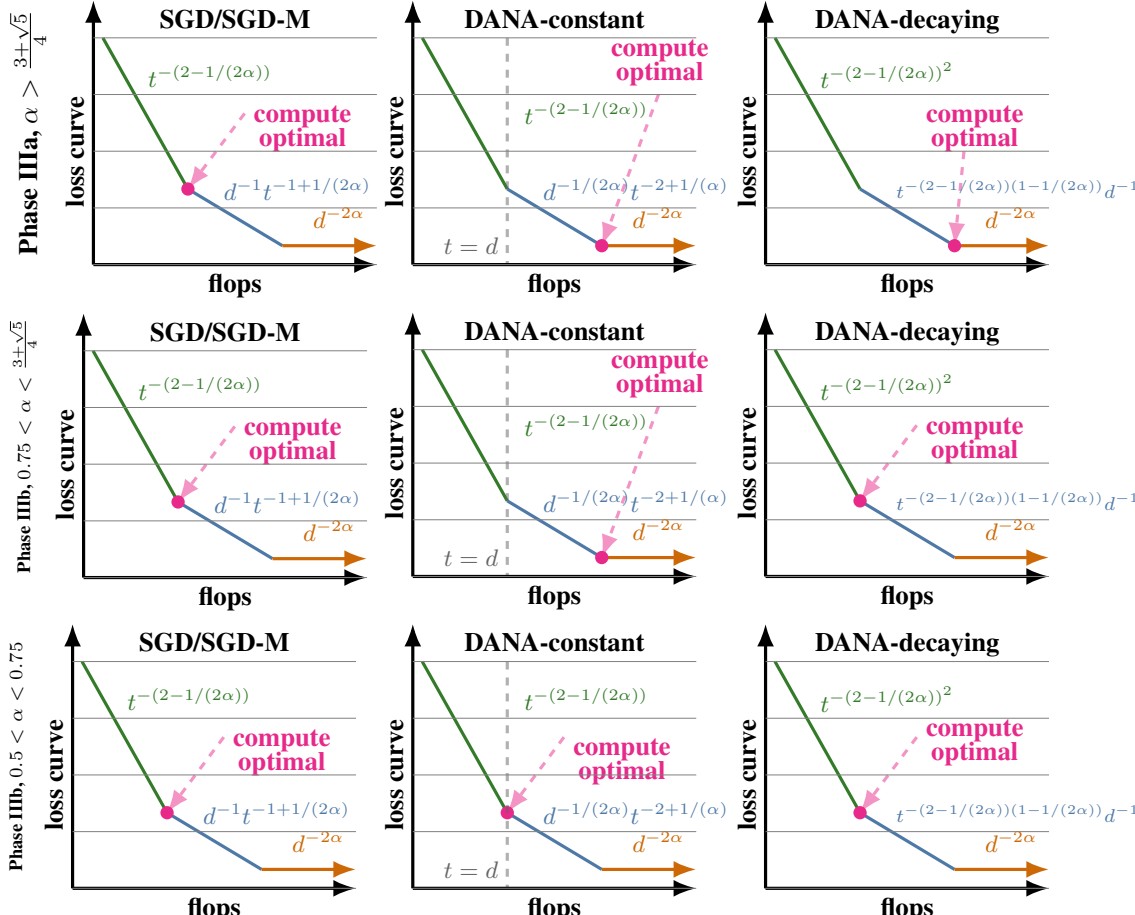

Figure 14: **Cartoon plots of the scaling laws for each algorithm above the high-dimensional line (Phase IIIa, IIIb).** Pictures for the scaling laws for all three algorithms in each of the different phases. When $t < d$, DANA-constant behaves like SGD/SGD-M. Observe that the trade off point for compute-optimum changes across phases and algorithms; indicated by (magenta). Variance, $\mathcal{K}_{pp}(t) =$ (green), embedding bias, $\mathcal{F}_{ac}(t) =$ (blue), and model capacity, $\mathcal{F}_0(t) =$ (orange). Even in the stochastic noise-dominated regime, acceleration occurs for DANA-constant and for DANA-decaying. We always use $\kappa_1 = \max\{0, 1 - 2\alpha\}$ and we use DANA-constant with $\kappa_2 = 1 + \kappa_1$ and $\kappa_3 = 0$ and DANA-decaying with $\kappa_3 = 1/(2\alpha)$, $\kappa_2 = \kappa_1$, and batch size $B = 1$.

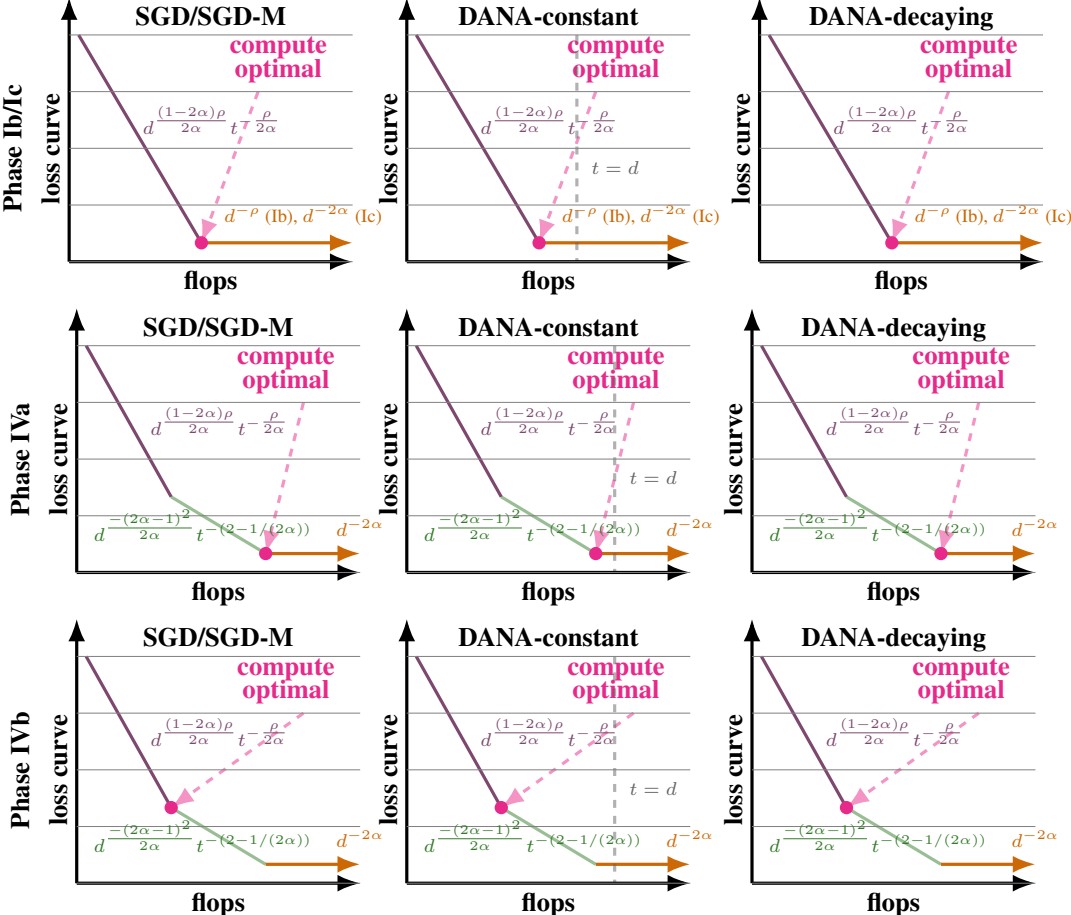

Figure 15: **Cartoon plots of the scaling laws for each algorithm below the high-dimensional line (Phase Ib/c, IVa/b).** Pictures for the scaling laws for all three algorithms in each of the different phases. Observe that the scaling laws are all the same for every algorithm; Tradeoff in compute-optimum indicated by (magenta). Variance, $\mathcal{K}_{pp}(t)$ = (green), population bias, $\mathcal{F}_{pp}(t)$ = (purple), and model capacity, $\mathcal{F}_0(t)$ = (orange). SGD/SGD-M and DANA-constant/DANA-decaying have the same scaling behavior. **Here** $\rho = 2\alpha + 2\beta - 1$. We always use $\kappa_1 = \max\{0, 1 - 2\alpha\}$ and we use DANA-constant with $\kappa_2 = 1 + \kappa_1$ and $\kappa_3 = 0$ and DANA-decaying with $\kappa_3 = 1/(2\alpha)$, $\kappa_2 = \kappa_1$, and batch size $B = 1$.

# F Stochastic gradient descent (SGD)

The ODEs in (22) precisely yield the same scaling laws and compute-optimal curves as previous works have shown for the stochastic gradient descent (SGD) algorithm with constant learning rate. Of particular interest is reproducing the results of [80] and extending them to include learning rate schedules.

## F.1 Volterra equation for SGD

When $\gamma_3 \equiv 0$, then the updates in (Gen-Mom-SGD) are precisely SGD with learning rate schedule $\gamma_2(t)$. Moreover, we see that the iterates generated by the "coin-flipping" algorithm when $\gamma_3 \equiv 0$ give the exact same iterates as SGD. That is, SGD is the "coin-flipping" algorithm with $\gamma_3 \equiv 0$.

When $\gamma_3 \equiv 0$, the ODE in (22) becomes

$$\frac{\mathrm{d}\nu}{\mathrm{d}t} = \Omega(t; \lambda_j) \times \nu(t; \lambda_j) + g(t; \lambda_j)$$

$$\text{where} \quad \Omega(t; \lambda_j) = \begin{pmatrix} -2\gamma_2(t)B\lambda_j + B(B+1)\gamma_2^2(t)\lambda_j^2 & 0 & 0 \\ \gamma_1^2(t)B(B+1)\lambda_j^2 & -2\Delta(t) + \Delta^2(t) & 2\gamma_1(t)B\lambda_j(1 - \Delta(t)) \\ \gamma_1(t)B\lambda_j & 0 & -\Delta(t) - \gamma_2(t)B\lambda_j \end{pmatrix}.$$

We observe that the $\nu(t; \lambda_j)_1 = \rho_j^2(t)$ for $j = 1, \dots d$ completely decouple from $\xi_j^2$ and $\chi_j$. Therefore, the ODE reduces to solving the following linear ODEs for $\rho_j^2$, $j = 1, \dots, d$

$$\frac{\mathrm{d}}{\mathrm{d}t}\rho_j^2(t) = \left[ -2\gamma_2(t)B\lambda_j + B(B+1)\gamma_2^2(t)\lambda_j^2 \right]\rho_j^2(t) + \gamma_2^2(t)\lambda_j B\mathbb{E}\left[\mathscr{P}(t) \,|\, W\right].$$

This is the same ODE that appeared in [80] with $\gamma_2(t)$ a constant. Moreover when $\lambda_j = 0$, then $\frac{d}{dt}\rho_j^2(t) = 0$ and so $\rho_j^2(t) = \rho_j^2(0)$ for all $t$.

For $\lambda_j > 0$, using Duhamel's principle on these 1-dimensional non-homogeneous linear ODEs, we can explicitly solve for the $\rho_j^2(t)$:

$$\rho_j^2(t) = \exp\left(A(t)\right)\rho_j^2(0) + \int_0^t \exp\left(A(t, \lambda_j) - A(s, \lambda_j)\right)\gamma_2^2(s)\lambda_j B\mathbb{E}\left[\mathscr{P}(s) \,|\, W\right] \mathrm{d}s,$$

where $A(t, \lambda) \overset{\text{def}}{=} -2B\Gamma_2(t)\lambda + B(B+1)\widehat{\Gamma}_2(t)\lambda^2$, $\Gamma_2(t) \overset{\text{def}}{=} \int_0^t \gamma_2(r) \, \mathrm{d}r$, $\widehat{\Gamma}_2(t) \overset{\text{def}}{=} \int_0^t \gamma_2^2(r) \, \mathrm{d}r$.

Alternatively, in terms of the matrix $\Phi_\lambda(t, s)$, we have for SGD

$$[\Phi_\lambda(t, s)]_{11} = \exp(A(t, \lambda) - A(s, \lambda)). \tag{79}$$

provided that $\lambda > 0$ and when $\lambda = 0$, $\Phi_0(t, s)_{11} \equiv 1$.

**Remark F.1.** *We do not need to compute $[\Phi_\lambda(t, s)]_{12}$ since $\gamma_1(t) \equiv 0$ so it does not effect the kernel function.*

## F.2 SGD with learning rate schedule

Using the results above, we have the Volterra equation for SGD.

> **Volterra equation for SGD with learning rate schedule $\gamma_2(t)$.** We can now give an explicit representation for the expected loss of SGD, that is, following the arguments in Section C, with $\theta_0 = \Theta_0 = 0$,
>
> $$\mathbb{E}\left[\mathscr{P}(t) \,|\, W\right] = \underbrace{\mathscr{F}(t)}_{\substack{\text{forcing func.} \\ \text{grad. descent}}} + \underbrace{\int_0^t \mathscr{K}_t(s) \times \mathbb{E}\left[\mathscr{P}(\Theta_s) \,|\, W\right] \mathrm{d}s}_{\text{SGD noise}}, \tag{80}$$

where, for any contour $\Gamma$ containing $\hat{K}$, we have

$$\mathscr{F}(t) \stackrel{\text{def}}{=} \frac{-1}{2\pi i} \oint_\Gamma \langle \mathscr{R}(z, \hat{K}), (D^{1/2}b)^{\otimes 2} \rangle \exp\left( -2B\Omega_2(t)z + B(B+1)\Omega_2^2(t)z^2 \right) \mathrm{d}z$$

$$\mathscr{K}_s(t) \stackrel{\text{def}}{=} \frac{-1}{2\pi i} \gamma_2^2(s)B$$

$$\times \mathrm{Tr}\left( \oint_\Gamma z^2 \exp\left( -2Bz(\Omega_2(t) - \Omega_2(s)) + B(B+1)z^2(\Omega_2^2(t) - \Omega_2^2(s)) \right) \right.$$

$$\left. \times \mathscr{R}(z, \hat{K}) \, \mathrm{d}z \right),$$

and $\quad \Omega_2(t) \stackrel{\text{def}}{=} \int_0^t \gamma_2(t') \, \mathrm{d}t' \quad$ and $\quad \Omega_2^2(t) \stackrel{\text{def}}{=} \int_0^t \gamma_2^2(t') \, \mathrm{d}t'.$

The Volterra equation using the deterministic equivalent for $\hat{K}$ immediately follows.

**Volterra equation for SGD with learning rate schedule $\gamma_2(t)$ under the deterministic equivalent.** We can now give an explicit representation for the deterministic equivalent of SGD, that is, following the arguments in Section C, with $\theta_0 = \Theta_0 = 0$,

$$\mathcal{P}(t) = \underbrace{\mathcal{F}(t)}_{\text{grad. descent}} + \underbrace{\int_0^t \mathcal{K}_s(t) \times \mathcal{P}(s) \, \mathrm{d}s}_{\text{SGD noise}}, \tag{81}$$

where the term $\mathcal{F}(t)$ is the *forcing func.* and

where, for any contour $\Gamma$ containing $[0, 1]$, we have

$$\mathcal{F}(t) \stackrel{\text{def}}{=} \frac{-1}{2\pi i} \oint_\Gamma \langle \mathcal{R}(z), (D^{1/2}b)^{\otimes 2} \rangle \exp\left( -2B\Gamma_2(t)z + B(B+1)\widehat{\Gamma}_2(t)z^2 \right) \mathrm{d}z$$

$$= \int_0^\infty \exp\left( -2B\Gamma_2(t)u + B(B+1)\widehat{\Gamma}_2(t)u^2 \right) \mu_{\mathcal{F}}(\mathrm{d}u)$$

$$\mathcal{K}_s(t) \stackrel{\text{def}}{=} \frac{-1}{2\pi i} \gamma_2^2(s)B$$

$$\times \mathrm{Tr}\left( \oint_\Gamma z^2 \exp\left( -2Bz(\Gamma_2(t) - \Gamma_2(s)) + B(B+1)z^2(\widehat{\Gamma}_2(t) - \widehat{\Gamma}_2(s)) \right) \right.$$

$$\left. \times \mathcal{R}(z) \, \mathrm{d}z \right)$$

$$= \gamma_2^2(s)B \int_0^\infty \exp\left( -2B\gamma_2 u(\Gamma_2(t) - \Gamma_2(s)) + B(B+1)u^2(\widehat{\Gamma}_2(t) - \widehat{\Gamma}_2(s)) \right)$$

$$\times \mu_{\mathcal{K}}(\mathrm{d}u),$$

and $\quad \Gamma_2(t) \stackrel{\text{def}}{=} \int_0^t \gamma_2(r) \, \mathrm{d}r \quad$ and $\quad \widehat{\Gamma}_2(t) \stackrel{\text{def}}{=} \int_0^t \gamma_2^2(r) \, \mathrm{d}r.$

**Remark F.2.** *The results above also hold when $\gamma_2(t)$ is a constant and they exactly reproduce the results in [80].*

### F.2.1 Simplifying the Volterra equation for SGD with learning rate schedule

We now aim to simplify the Volterra equation for SGD and to do so, we will assume that the SGD learning rate schedule is decaying.

**Assumption 4** (SGD Learning Rate). *Suppose the learning rate schedule $\gamma_2 : \mathbb{R}_{\geq 0} \to \mathbb{R}_{>0}$ is a nonincreasing function. Moreover, we let $\bar{\gamma}_2 \stackrel{\text{def}}{=} \gamma_2(0)$, that is $\gamma_2(t) \leq \bar{\gamma}_2$ for all t.*

Throughout this section, we assume that Assumption 4 holds. Prior to this section, i.e., the derivation of the Volterra equation (81), holds regardless of Assumption 4.

Next, we need to introduce upper and lower bounds on the SGD kernel. For this, we introduce two kernel functions,

$$\overline{\mathcal{K}}^{\text{SGD}}(t) \stackrel{\text{def}}{=} \int_0^1 \exp(-2But + B(B+1)\bar{\gamma}_2 u^2 t) \, \mu_{\mathcal{K}}(du)$$

$$\text{and} \quad \underline{\mathcal{K}}^{\text{SGD}}(t) \stackrel{\text{def}}{=} \int_0^1 \exp(-2But) \, \mu_{\mathcal{K}}(du). \tag{82}$$

To simplify the notation, we write $\overline{\mathcal{K}}(t) \stackrel{\text{def}}{=} \overline{\mathcal{K}}^{\text{SGD}}(t)$ and $\underline{\mathcal{K}}(t) \stackrel{\text{def}}{=} \underline{\mathcal{K}}^{\text{SGD}}(t)$. Next, we know that $\widehat{\Gamma}_2(t) - \widehat{\Gamma}_2(s) \leq \bar{\gamma}_2(\Gamma_2(t) - \Gamma_2(s))$. Using this observation, we have, for any $0 \leq s \leq t$

$$\gamma_2^2(s)B \times \underline{\mathcal{K}}(\Gamma_2(t) - \Gamma_2(s)) \leq \mathcal{K}_s(t) \leq \gamma_2^2(s)B \times \overline{\mathcal{K}}(\Gamma_2(t) - \Gamma_2(s)). \tag{83}$$

Similarly, we define an upper and lower bound on the forcing function $\mathcal{F}$

$$\overline{\mathcal{F}}(t) \stackrel{\text{def}}{=} \int_0^1 \exp(-2But + B(B+1)\bar{\gamma}_2 u^2 t) \, \mu_{\mathcal{F}}(du) \quad \text{and} \quad \underline{\mathcal{F}}(t) \stackrel{\text{def}}{=} \int_0^1 \exp(-2But) \, \mu_{\mathcal{F}}(du). \tag{84}$$

It is clear that

$$\underline{\mathcal{F}}(\Gamma_2(t)) \leq \mathcal{F}(t) \leq \overline{\mathcal{F}}(\Gamma_2(t)).$$

In the following lemma, we show that $\overline{\mathcal{K}}$ and $\overline{\mathcal{F}}$ are monotonically decreasing in $t$.

**Lemma F.1** ($\overline{\mathcal{K}}$ and $\overline{\mathcal{F}}$ are decreasing functions)**.** *Suppose Assumption 4 holds and the learning rate satisfies*

$$\frac{2}{B+1} > \bar{\gamma}_2.$$

*Then $\overline{\mathcal{F}}$ and $\overline{\mathcal{K}}$ are decreasing functions in t. In particular, the function $\mathcal{F}$ is bounded.*

*Proof.* First, if $\overline{\mathcal{F}}$ is decreasing, then it is clear that $\mathcal{F}$ is bounded. Therefore it only remains to show that $\overline{F}$ and $\overline{K}$ are decreasing. This amounts to showing that

$$u(-2Bt + B(B+1)\bar{\gamma}_2 u) < 0, \quad \text{for all } u \in [0, 1].$$

This holds provided that $\frac{2}{B+1} > \bar{\gamma}_2$. $\qquad \square$

Lastly, we introduce *Kesten constant* for the kernel of SGD as the following

$$\|\overline{\mathcal{K}}\| \stackrel{\text{def}}{=} \int_0^\infty \overline{\mathcal{K}}(t) \, dt. \tag{85}$$

We now proceed to simplify the Volterra equation for SGD in (81) using a result similar to Lemma D.3 in [27].

**Lemma F.2.** *Suppose Assumption 4 holds, that is, $\gamma_2(t)$ is a nonincreasing learning rate schedule and $\frac{2}{B+1} > \bar{\gamma}_2$. Let $\Gamma_2(t) = \int_0^t \gamma_2(r) \, dr$. Then for all $t \geq 0$,*

$$\mathcal{P}(t) \geq \mathcal{F}(t) + \int_0^t \gamma_2^2(s)B \times \underline{\mathcal{K}}(\Gamma_2(t) - \Gamma_2(s))\mathcal{F}(s) \, ds.$$

*If, in addition, there exists a $\varepsilon > 0$ and $T(\varepsilon) > 0$,*

$$\int_0^t \overline{\mathcal{K}}(s)\overline{\mathcal{K}}(t - s) \, ds \leq 2(1+\varepsilon)\|\overline{\mathcal{K}}\|\overline{\mathcal{K}}(t) \quad \text{and} \quad 2\bar{\gamma}_2 B\|\overline{\mathcal{K}}\|(1+\varepsilon) < 1, \tag{86}$$

*then for all t,*

$$\mathcal{P}(t) \leq \mathcal{F}(t) + C \int_0^t \gamma_2^2(s)B \times \overline{\mathcal{K}}(\Gamma_2(t) - \Gamma_2(s))\mathcal{F}(s) \, ds$$

*for*

$$C = \left( \frac{\overline{\mathcal{K}}(0)}{\overline{\mathcal{K}}(T)(2\varepsilon + 1)} + 1 \right) \frac{1}{1 - 2\bar{\gamma}_2 B\|\overline{\mathcal{K}}\|(1+\varepsilon)}.$$

*Proof.* We recall that

$$\mathcal{P}(t) = \mathcal{F}(t) + \int_0^t \mathcal{K}_s(t) \times \mathcal{P}(s) \, \mathrm{d}s.$$

The lower bound holds trivially after noting that $\mathcal{K}_s(t) \geq \gamma_2^2(s) B \times \underline{\mathcal{K}}(\Gamma_2(t) - \Gamma_2(s))$ and $\mathcal{P}(s) \geq \mathcal{F}(s)$. For the upper bound, we start with the following. Let us define the convolution map

$$\mathcal{G}(\mathcal{F})(t) \stackrel{\text{def}}{=} \int_0^t \mathcal{K}_s(t) \mathcal{F}(s) \, \mathrm{d}s,$$

with the composition of the convolution mapping by

$$\mathcal{G}^2(\mathcal{F})(t) = \int_0^t \mathcal{K}_s(t) \mathcal{G}(\mathcal{F})(s) \, \mathrm{d}s = \int_0^t \int_0^s \mathcal{K}_s(t) \mathcal{K}_r(s) \mathcal{F}(r) \, \mathrm{d}r \, \mathrm{d}s.$$

This naturally extends to $\mathcal{G}^j(\mathcal{F})(t)$. Next, let us define two functions $h \stackrel{\text{def}}{=} \mathcal{F} \circ \Gamma_2^{-1}$ and $g(u) \stackrel{\text{def}}{=} B \times \gamma_2(\Gamma_2^{-1}(u))$ where $\Gamma_2(t) = \int_0^t \gamma_2(s) \, \mathrm{d}s$. Moreover, we introduce another convolution mapping given by

$$\overline{\mathcal{G}}(h)(t) \stackrel{\text{def}}{=} \int_0^t \overline{\mathcal{K}}(t - u) g(u) h(u) \, \mathrm{d}u,$$

where the composition is given by

$$\overline{\mathcal{G}}^2(h)(t) = \int_0^t \overline{\mathcal{K}}(t - s) g(s) \overline{\mathcal{G}}(h)(s) \, \mathrm{d}s = \int_0^t \int_0^s \overline{\mathcal{K}}(t - s) g(s) \overline{\mathcal{K}}(s - u) g(u) h(u) \, \mathrm{d}u.$$

Again we extend this to $j$ compositions, $\overline{\mathcal{G}}^j(h)(t)$.

As the kernel function $\mathcal{K}_s(t)$ and forcing function $\mathcal{F}(t)$ are non-negative, we have that

$$\mathcal{P}(t) = \mathcal{F}(t) + \sum_{j=1}^{\infty} \mathcal{G}^j(\mathcal{F})(t). \tag{87}$$

Next, we prove the following claim. *Claim: $\mathcal{G}^j(\mathcal{F})(t) \leq \overline{\mathcal{G}}^j(h)(\Gamma_2(t))$ for all $j \geq 1$.*

*Proof of Claim:* To see this, we will do the case when $j = 3$ in detail, but the idea will extend to all $j \geq 1$. For this, we have that

$$\overline{\mathcal{G}}^3(h)(\Gamma_2(t)) = \int_0^{\Gamma_2(t)} \int_0^w \int_0^u \overline{\mathcal{K}}(\Gamma_2(t) - w) \overline{\mathcal{K}}(w - u) \overline{\mathcal{K}}(u - v) g(w) g(u) g(v) h(v) \, \mathrm{d}v \, \mathrm{d}u \, \mathrm{d}w.$$

We consider the change of variables $v = \Gamma(p)$ where $dv = \gamma_2(p) \, \mathrm{d}p$. Then

$$\overline{\mathcal{G}}^3(h)(\Gamma_2(t)) = \int_0^{\Gamma_2(t)} \int_0^w \int_0^{\Gamma^{-1}(u)} \overline{\mathcal{K}}(\Gamma_2(t) - w) \overline{\mathcal{K}}(w - u) \overline{\mathcal{K}}(u - \Gamma_2(p))$$
$$\times g(w) g(u) B \gamma_2^2(p) \mathcal{F}(p) \, \mathrm{d}p \, \mathrm{d}u \, \mathrm{d}w.$$

Now consider the change of variables where $u = \Gamma_2(r)$. Then,

$$\overline{\mathcal{G}}^3(h)(\Gamma_2(t)) = \int_0^{\Gamma_2(t)} \int_0^w \int_0^{\Gamma^{-1}(u)} \overline{\mathcal{K}}(\Gamma_2(t) - w) \overline{\mathcal{K}}(w - u) \overline{\mathcal{K}}(u - \Gamma_2(p))$$
$$\times g(w) g(u) B \gamma_2^2(p) \mathcal{F}(p) \, \mathrm{d}p \, \mathrm{d}u \, \mathrm{d}w$$
$$= \int_0^{\Gamma_2(t)} \int_0^{\Gamma_2^{-1}(w)} \int_0^r \overline{\mathcal{K}}(\Gamma_2(t) - w) \overline{\mathcal{K}}(w - \Gamma_2(r)) \overline{\mathcal{K}}(\Gamma_2(r) - \Gamma_2(p))$$
$$\times g(w) \gamma_2^2(r) B^2 \gamma_2^2(p) \mathcal{F}(p) \, \mathrm{d}p \, \mathrm{d}r \, \mathrm{d}w$$
$$= \int_0^t \int_0^s \int_0^r \overline{\mathcal{K}}(\Gamma_2(t) - \Gamma_2(s)) \overline{\mathcal{K}}(\Gamma_2(s) - \Gamma_2(r)) \overline{\mathcal{K}}(\Gamma_2(r) - \Gamma_2(p))$$
$$\times \gamma_2^2(s) \gamma_2^2(r) B^3 \gamma_2^2(p) \mathcal{F}(p) \, \mathrm{d}p \, \mathrm{d}r \, \mathrm{d}s.$$

The last equality follows from the change of variables $w = \Gamma_2(s)$. The result immediately follows since $\overline{\mathcal{K}}(\Gamma_2(t) - \Gamma_2(s))\gamma_2^2(s)B \geq \mathcal{K}_s(t)$ and the kernels are nonnegative. This proves the claim.

Therefore, we have from (87)

$$\mathcal{P}(t) \leq \mathcal{F}(t) + \sum_{j=1}^{\infty} \overline{\mathcal{G}}^j(h)(\Gamma_2(t)).$$

Next, we show that the map $\overline{\mathcal{G}}(h)$ is contracting and in particular,

$$\overline{\mathcal{G}}^2(h)(t) = \int_0^t \overline{\mathcal{K}}(t-s)g(s)\overline{\mathcal{G}}(h)(s) \, \mathrm{d}s = \int_0^t \int_0^s \overline{\mathcal{K}}(t-s)g(s)\overline{\mathcal{K}}(s-u)g(u)h(u) \, \mathrm{d}u \, \mathrm{d}s$$

$$= \int_0^t \left( \int_u^t \overline{\mathcal{K}}(t-s)\overline{\mathcal{K}}(s-u)g(s) \, \mathrm{d}s \right) g(u)h(u) \, \mathrm{d}u$$

$$\leq \int_0^t \overline{\mathcal{K}}^{*2}(t-u)g(u)^2 h(u) \, \mathrm{d}u,$$

where the third equality is since $u < s < t$. The last transition is by change of variables and the assumption that $\gamma_2(t)$ is a nonincreasing function. Consecutive application of the convolution map will then yield by induction,

$$\overline{\mathcal{G}}^j(h)(t) \leq \int_0^t \overline{\mathcal{K}}^{*j}(t-u)g(u)^j h(u) \, \mathrm{d}u.$$

Therefore, expanding the loss and using the upper bound, and denote $q = 2\bar{\gamma}_2 B(1+\varepsilon)\|\overline{\mathcal{K}}\|$ such that $q < 1$,

$$\mathcal{P}(t) \leq \mathcal{F}(t) + \sum_{j=1}^{\infty} \overline{\mathcal{G}}^j(h)(\Gamma_2(t))$$

$$\leq \mathcal{F}(t) + \sum_{j=1}^{\infty} \int_0^{\Gamma_2(t)} \overline{\mathcal{K}}^{*j}(\Gamma_2(t) - u)g(u)^j h(u) \, \mathrm{d}u$$

$$\leq \mathcal{F}(t) + \left( \sum_{j=1}^{\infty} (2\bar{\gamma}_2 B\|\overline{\mathcal{K}}\|(1+\varepsilon))^{j-1} \right) C_1 \int_0^{\Gamma_2(t)} \overline{\mathcal{K}}(\Gamma_2(t) - u)g(u)h(u) \, \mathrm{d}u$$

$$\leq \mathcal{F}(t) + \left( \frac{1}{1-q} \right) \times C_1 \times \int_0^t B\gamma_2^2(s) \times \overline{\mathcal{K}}(\Gamma_2(t) - \Gamma_2(s))\mathcal{F}(s) \, \mathrm{d}s.$$

The third transition follows from Lemma F.3 with $C_1 = \frac{\overline{\mathcal{K}}(0)}{\overline{\mathcal{K}}(T)(2\varepsilon+1)} + 1$ and the last transition follows from a change of variables $u = \Gamma_2(s)$. $\qquad\square$

**Lemma F.3** (Lemma D.4 in [27] and Lemma IV.4.7 in [8]). *Suppose $\bar{\gamma}_2 < \frac{2}{B+1}$, i.e., $\overline{\mathcal{K}}$ is monotonically decreasing. Suppose additionally that $\|\overline{\mathcal{K}}\| < \infty$ and for some $\varepsilon > 0$, there exists a $T(\varepsilon) > 0$ such that*

$$\int_0^t \overline{\mathcal{K}}(s)\overline{\mathcal{K}}(t-s) \leq 2(1+\varepsilon)\|\overline{\mathcal{K}}\|\overline{\mathcal{K}}(t), \quad \text{for all } t \geq T.$$

*Then for all $n \geq 0$*

$$\sup_t \left\{ \frac{\overline{\mathcal{K}}^{*n}(t)}{\overline{\mathcal{K}}(t)} \right\} \leq (2\|\overline{\mathcal{K}}\|(1+\varepsilon))^{n-1} \left( \frac{\overline{\mathcal{K}}(0)}{\overline{\mathcal{K}}(T)(2\varepsilon+1)} + 1 \right).$$

We immediately get a corollary of Lemma F.2.

**Corollary F.1.** *Under the assumptions of Lemma F.2, the following holds*

$$\underline{\mathcal{F}}(t) + \int_0^t \gamma_2^2(s)B \times \underline{\mathcal{K}}(\Gamma_2(t) - \Gamma_2(s))\underline{\mathcal{F}}(s) \, \mathrm{d}s \leq \mathcal{P}(t)$$

$$\leq \overline{\mathcal{F}}(t) + C \times \int_0^t \gamma_2^2(s)B \times \overline{\mathcal{K}}(\Gamma_2(t) - \Gamma_2(s))\overline{\mathcal{F}}(s) \, \mathrm{d}s,$$

*where $C$ is the constant in Lemma F.2.*

The goal will be to show that the lower and upper bound on $\mathcal{P}(t)$ from Corollary F.1 have the same asymptotics (see Section F.2.2 for forcing function and Section F.2.3). Moreover, using the results from [80], we know that $\overline{\mathcal{K}}$ satisfies the Kesten's lemma condition (86).

**Proposition F.1** (SGD and Kesten's condition, Proposition G.2 [80])**.** *Suppose* $\alpha > \frac{1}{4}$. *For any* $\varepsilon > 0$, *there is an $M$ sufficiently large so that for $Bt \in [M, d^{2\alpha}/M]$,*

$$\int_0^t \overline{\mathcal{K}}(s)\overline{\mathcal{K}}(t-s) \leq (2+\varepsilon)\|\overline{\mathcal{K}}\|\overline{\mathcal{K}}(t).$$

Moreover, we have estimates on the Kesten constant $\|\overline{\mathcal{K}}\|$.

**Lemma F.4** (Boundedness of $\|\overline{\mathcal{K}}\|$, Corollary G.1 [80])**.** *When $2\alpha > 1$ and $\overline{\gamma}_2(B+1) < 2$,*

$$\|\overline{\mathcal{K}}\| = \frac{1}{2B}\sum_{j=1}^{\infty} \frac{j^{-2\alpha}}{1 - \frac{1}{2}\overline{\gamma}_2(B+1)}(1 + o(1)).$$

*When $2\alpha < 1$,*

$$\|\overline{\mathcal{K}}\| = \frac{1}{2B}\frac{v^{1-2\alpha}}{1-2\alpha}(1 + o(1)).$$

We remark that $\|\overline{\mathcal{K}}\|$ is approximately $\frac{1}{2B}\text{Tr}(D)$; here $\text{Tr}(D) = \sum_{j=1}^{d} j^{-2\alpha}$. This leads to the main result.

**Proposition F.2.** *Let $v, d$ be admissible. When $2\alpha > 1$, suppose*

$$\overline{\gamma}_2(B+1) < 2 \quad and \quad \overline{\gamma}_2 < \frac{1}{\sum_{j=1}^{\infty} j^{-2\alpha}}.$$

*Or when $2\alpha < 1$, suppose $\overline{\gamma}_2 = c \times d^{2\alpha-1}$ where $c > 0$ is any constant and*

$$\overline{\gamma}_2(B+1) < 2 \quad and \quad \frac{c}{1-2\alpha} < 1.$$

*Then the expected loss $\mathcal{P}$ is bounded and*

$$\underline{\mathcal{F}}(\Gamma_2(t)) + \int_0^t \gamma_2^2(s)B \times \underline{\mathcal{K}}(\Gamma_2(t) - \Gamma_2(s))\underline{\mathcal{F}}(\Gamma_2(s)) \, ds \leq \mathcal{P}(t)$$

$$\leq \overline{\mathcal{F}}(\Gamma_2(t)) + C \times \int_0^t \gamma_2^2(s)B \times \overline{\mathcal{K}}(\Gamma_2(t) - \Gamma_2(s))\overline{\mathcal{F}}(\Gamma_2(s)),$$

*where $C$ is the constant in Lemma F.2.*

*Proof.* The proof combines the previous results with the idea that the conditions on the learning rates ensure that $\overline{\mathcal{F}}$ is bounded and $2B\overline{\gamma}_2\|\mathcal{K}\| < 1$. $\qquad\square$

### F.2.2 Forcing function for SGD with decaying learning rate schedule

In light of Corollary F.1, we can now define the different components of the forcing function. Following the outline in Section D.1, we have three components for the upper forcing function $\overline{\mathcal{F}}$ and the lower forcing function $\underline{\mathcal{F}}$,

$$\underline{\mathcal{F}}_{pp}(t) \stackrel{\text{def}}{=} \frac{1}{2\alpha}\int_0^1 \exp(-2But) \times u^{(2\beta-1)/(2\alpha)} \, du$$

$$\overline{\mathcal{F}}_{pp}(t) \stackrel{\text{def}}{=} \frac{1}{2\alpha}\int_0^1 \exp(-2But + B(B+1)\overline{\gamma}_2 u^2 t) \times u^{(2\beta-1)/(2\alpha)} \, du$$

$$\underline{\mathcal{F}}_{ac}(t) \stackrel{\text{def}}{=} \frac{c_\beta}{2\alpha d}\int_0^1 \exp(-2But) \times u^{-1/(2\alpha)} \, du$$

$$\overline{\mathcal{F}}_{ac}(t) \stackrel{\text{def}}{=} \frac{c_\beta}{2\alpha d}\int_0^1 \exp(-2But + B(B+1)\overline{\gamma}_2 u^2 t) \times u^{(2\beta-1)/(2\alpha)} \, du,$$

where $c_\beta \overset{\text{def}}{=} \sum_{j=1}^\infty j^{-2\beta}$ if $\beta > \frac{1}{2}$ and 0 otherwise, and lastly,

$$\underline{\mathcal{F}}_0(t), \overline{\mathcal{F}}_0(t) \overset{\text{def}}{=} \sum_{j=1}^v \frac{j^{-2\alpha-2\beta}}{1 + j^{-2\alpha}d^{2\alpha}\kappa(v/d)} \quad \text{where } \kappa > 0 \text{ is the unique solution of } \int_0^{v/d} \frac{\kappa \, dx}{\kappa + x^{2\alpha}} = 1.$$

For the pure point terms, we get the following proposition.

**Proposition F.3** (Pure point forcing term, Proposition H.2 [80]). *Suppose $2\alpha + 2\beta > 1$. For any $\varepsilon > 0$, there is an $M > 0$ so that for $Bt \geq M$,*

$$|\overline{\mathcal{F}}_{pp}(t) - g(t)| \leq \varepsilon \times g(t) \quad \text{and} \quad |\underline{\mathcal{F}}_{pp}(t) - g(t)| \leq \varepsilon \times g(t)$$

*where*

$$g(t) \overset{\text{def}}{=} (2\alpha)^{-1}(2B)^{1/(2\alpha)-\beta/\alpha-1} \times \Gamma(\tfrac{\beta}{\alpha} - \tfrac{1}{2\alpha} + 1) \times t^{-(1+\beta/\alpha)+1/(2\alpha)}.$$

*Furthermore, for any $\tilde{M} > 0$, there exists some constants $C, \tilde{C}, c > 0$ independent of $d$ so that*

$$c \leq \overline{\mathcal{F}}_{pp}(t), \underline{\mathcal{F}}_{pp}(t) \leq C \quad \text{if } Bt < \tilde{M}$$

*and if $t > \tilde{M}d^{2\alpha}$*

$$\overline{\mathcal{F}}_{pp}(t), \underline{\mathcal{F}}_{pp}(t) \leq \tilde{C} \times \overline{\mathcal{F}}_0(t).$$

As for $\mathcal{F}_0$, we have the following proposition.

**Proposition F.4** (Asymptotic for $\mathcal{F}_0$, Proposition H.3 [80]). *Suppose $v$ and $d$ are admissible such that the ratio $v/d > 1$ and suppose $2\alpha + 2\beta > 1$. Let $0 < \kappa(v/d) < \infty$ be the unique solution to*

$$1 = \int_0^{v/d} \frac{\kappa}{\kappa + u^{2\alpha}} \, du.$$

*Then as $d \to \infty$*

$$\overline{\mathcal{F}}_0(t) \, \underline{\mathcal{F}}_0(t) \sim \begin{cases} \frac{d^{-2\alpha}}{\kappa} \big( \sum_{j=1}^v j^{-2\beta} \big), & \text{if } 2\beta > 1 \\ d^{1-2(\alpha+\beta)} \int_0^{v/d} \frac{u^{-2\beta}}{\kappa + u^{2\alpha}} \, du, & \text{if } 2\beta < 1. \end{cases}$$

Lastly, we have a proposition for the absolutely continuous part of $\mathcal{F}_{ac}(t)$.

**Proposition F.5** (Absolutely continuous forcing function, Proposition H.4 [80]). *There exists a constant $C(\alpha, \beta) > 0$ such that*

$$\overline{\mathcal{F}}_{ac}(t), \underline{\mathcal{F}}_{ac}(t) \leq \begin{cases} C \times \overline{\mathcal{F}}_0(t), & \text{if } 2\beta > 1, \, 2\alpha < 1 \\ 0, & \text{if } 2\beta < 1. \end{cases}$$

*Suppose now $2\alpha > 1$ and $2\beta > 1$. For any $\varepsilon > 0$, there is an $M > 0$ so that for $Bt \in [M, d^{2\alpha}/M]$,*

$$|\overline{\mathcal{F}}_{ac}(t) - g(t)| \leq \varepsilon \times g(t) \quad \text{and} \quad |\underline{\mathcal{F}}_{ac}(t) - g(t)| \leq \varepsilon \times g(t)$$

$$\text{where} \quad g(t) \overset{\text{def}}{=} \big( \sum_{j=1}^v j^{-2\beta} \big)(2B)^{-1+1/(2\alpha)}(2\alpha)^{-1}\Gamma(1 - \tfrac{1}{2\alpha})t^{-1+1/(2\alpha)} \times d^{-1}.$$

*Furthermore, for any $\tilde{M} > 0$, there exists some constants $C, c > 0$ independent of $d$ so that*

$$\overline{\mathcal{F}}_{ac}(t), \underline{\mathcal{F}}_{ac}(t) \leq \begin{cases} C \times d^{-1}, & \text{if } Bt \leq \tilde{M} \\ c \times \overline{\mathcal{F}}_0(t), & \text{if } Bt \geq \tilde{M}d^{2\alpha}. \end{cases}$$

Combining all these propositions, we have the following conclusion.

**Proposition F.6** (Forcing function for SGD, Corollary F.1 [80]). *For any $\alpha, \beta$ with $\alpha, \beta \neq \frac{1}{2}$ and $\alpha + \beta > \frac{1}{2}$ there is a function $C(t)$ bounded above for all $t$ so that*

$$\frac{1}{C(t)}\big(\underline{\mathcal{F}}_{pp}(\Gamma_2(t)) + \underline{\mathcal{F}}_{ac}(\Gamma_2(t)) + \underline{\mathcal{F}}_0(\Gamma_2(t))\big) \leq \underline{\mathcal{F}}(\Gamma_2(t)) \leq \mathcal{F}(t)$$

$$\leq \overline{\mathcal{F}}(\Gamma_2(t)) \leq C(t)\big(\overline{\mathcal{F}}_{pp}(\Gamma_2(t)) + \overline{\mathcal{F}}_{ac}(\Gamma_2(t)) + \overline{\mathcal{F}}_0(\Gamma_2(t))\big).$$

*Moreover, for any $\varepsilon > 0$, there is a $M(\varepsilon)$ large enough that $C(t) \leq 1 + \varepsilon$ for $\Gamma_2(t)B \in [M, d^{2\alpha}/M]$ and for $\Gamma_2(t)B > Md^{2\alpha}$. Lastly, the upper and lower bounds satisfy*

$$\underline{\mathcal{F}}_{pp}(t) \sim \overline{\mathcal{F}}_{pp}(t), \quad \underline{\mathcal{F}}_{ac}(t) \sim \overline{\mathcal{F}}_{ac}(t), \quad \text{and} \quad \underline{\mathcal{F}}_0(t) \sim \overline{\mathcal{F}}_0(t).$$

| SGD-M | SGD |
|---|---|
| $\mathcal{F}_0(t) \asymp d^{-2\alpha+\max\{0,1-2\beta\}}$ | $\mathcal{F}_0(t) \asymp d^{-2\alpha+\max\{0,1-2\beta\}}$ |
| $\mathcal{F}_{pp}(t) \asymp (\gamma_{\mathrm{eff}}Bt)^{-1-\frac{2\beta-1}{2\alpha}}$ | $\mathcal{F}_{pp}(t) \asymp (\gamma_2 Bt)^{-(1+\beta/\alpha)+1/(2\alpha)}$ |
| $\mathcal{F}_{ac}(t) \leq \begin{cases} C \times \mathcal{F}_0(t), & \text{if } 2\beta > 1, 2\alpha < 1 \\ 0, & \text{if } 2\beta < 1 \end{cases}$ 
 if $2\beta > 1, 2\alpha > 1, \mathcal{F}_{ac} \asymp d^{-1}(\gamma_{\mathrm{eff}}Bt)^{-1+1/(2\alpha)}$ | $\mathcal{F}_{ac}(t) \asymp \begin{cases} C \times \mathcal{F}_0(t), & \text{if } 2\beta > 1, 2\alpha > 1 \\ 0, & \text{if } 2\beta < 1. \end{cases}$ 
 if $2\beta > 1, 2\alpha > 1, \mathcal{F}_{ac}(t) \asymp d^{-1}(\gamma_2 Bt)^{-1+1/(2\alpha)}$ |
| $\mathcal{K}_{pp}(t) \asymp \gamma_{\mathrm{eff}}^2 B (\gamma_{\mathrm{eff}}Bt)^{-2+1/(2\alpha)}$ | $\mathcal{K}_{pp}(t) \asymp \gamma_2^2 B(\gamma_2 Bt)^{-2+1/(2\alpha)}$ |

Table 8: **Asymptotics for the forcing and kernel functions for constant learning rate SGD/SGD-M.** See Section G/Section F for details/proofs of the derivations for these asymptotics of SGD-M/SGD. Here the constant $C$ is independent of dimension and $\gamma_{\mathrm{eff}} = \gamma_2 + \frac{\gamma_3}{\delta}$.

### F.2.3 Kernel function for SGD with decaying learning rate schedule

We perform a similar analysis for the kernel function for SGD. To do so, we introduce two (pure point) kernel functions:

$$\overline{\mathcal{K}}_{pp}(t) \overset{\text{def}}{=} \frac{1}{2\alpha} \int_0^1 \exp(-2But + B(B+1)\bar{\gamma}_2 u^2 t) \times u^{1-1/(2\alpha)} \, du$$

$$\text{and} \quad \underline{\mathcal{K}}_{pp}(t) \overset{\text{def}}{=} \frac{1}{2\alpha} \int_0^1 \exp(-2But) \times u^{1-1/(2\alpha)} \, du.$$

The first result states that the upper and lower $\mathcal{K}_{pp}$ are asymptotically the same.

**Proposition F.7** ($\mathcal{K}_{pp}$ asymptotic, Proposition H.5 [80]). *Suppose $\alpha > 1/4$. For any $\varepsilon > 0$, there is an $M > 0$ so that for $Bt \geq M$,*

$$|\overline{\mathcal{K}}_{pp}(t) - g(t)| \leq \varepsilon \times g(t) \quad \text{and} \quad |\underline{\mathcal{K}}_{pp}(t) - g(t)| \leq \varepsilon \times g(t)$$

*where*

$$g(t) \overset{\text{def}}{=} (2\alpha)^{-1}(2B)^{-2+1/(2\alpha)} \times \Gamma\left(2 - \tfrac{1}{2\alpha}\right) \times t^{-2+1/(2\alpha)}.$$

*Moreover, for any $\tilde{M} > 0$, there exists constants $c, C, \tilde{C} > 0$, such that when $2\alpha > 1$,*

$$c \leq \overline{\mathcal{K}}_{pp}(t) \leq C, \quad c \leq \underline{\mathcal{K}}_{pp}(t) \leq C, \quad \text{if } Bt \leq \tilde{M}$$

*and when $2\alpha < 1$,*

$$\overline{\mathcal{K}}_{pp}(t) \leq \hat{C} \times d^{2\alpha-1} \quad \underline{\mathcal{K}}_{pp}(t) \leq \hat{C} \times d^{2\alpha-1}, \quad \text{if } Bt \leq \tilde{M}.$$

*Furthermore, for any $\tilde{M} > 0$, there exists a constant $\tilde{C} > 0$, such that*

$$\overline{\mathcal{K}}_{pp}(t) \leq \tilde{C} \times \overline{\mathcal{F}}_0(t), \quad \underline{\mathcal{K}}_{pp}(t) \leq \tilde{C} \times \underline{\mathcal{F}}_0(t) \quad \text{if } Bt \geq \tilde{M}d^{2\alpha}.$$

This directly leads to the main result for the kernel function.

**Proposition F.8** (Kernel estimation, Proposition G.1 [80]). *Suppose $\alpha > 1/4$. There is a positive function $C(t)$ so that*

$$\frac{1}{C(t)} \times \gamma_2^2(s)B \times \underline{\mathcal{K}}_{pp}(\Gamma_2(t) - \Gamma_2(s)) \leq \mathcal{K}_s(t) \leq C(t) \times \gamma_2^2(s)B \times \overline{\mathcal{K}}_{pp}(\Gamma_2(t) - \Gamma_2(s)),$$

*and $C(t)$ is bounded independent of $d$ by a function of $M$ for all $\Gamma_2(t)B < d^{2\alpha}M$. Moreover for any $\varepsilon > 0$ there is an $M$ sufficiently large so that for $\Gamma_2(t)B \in [M, d^{2\alpha}/M], C(t) < 1 + \varepsilon$. Lastly, the upper and lower bounds satisfy*

$$\overline{\mathcal{K}}_{pp}(t) \sim \underline{\mathcal{K}}_{pp}(t).$$

Propositions F.4, F.5, F.3, and F.6 for the forcing function and their companions, Propositions F.7 and F.8 for the kernel function, allow us to simplify the Volterra equation.

**Proposition F.9** (Simplification of Volterra equation for SGD with decreasing learning rates). *Suppose* $\gamma_2(t)$ *is a decreasing learning rate schedule (i.e., Assumption 4 holds) and suppose the assumptions of Proposition F.2 hold. Let* $\alpha > 1/4$. *First suppose that the forcing function with the decaying learning rate is integrable, that is,*

$$\overline{\mathcal{C}} \overset{def}{=} \overline{\mathcal{C}}(\gamma_2, B) \overset{def}{=} \int_0^\infty h(\Gamma_2(s))\gamma_2^2(s) \ ds < \infty,$$

*where* $\quad h(t) \overset{def}{=} (2\alpha)^{-1}\Gamma(\frac{\beta}{\alpha} - \frac{1}{2\alpha} + 1) \times (2Bt)^{-(1+\beta/\alpha)+1/(2\alpha)} + c_\beta \times (2Bt)^{-1+1/(2\alpha)} \times d^{-1}.$
(88)

*Here* $c_\beta \overset{def}{=} \left( \sum_{j=1}^v j^{-2\beta} \right)(2\alpha)^{-1}\Gamma(1 - \frac{1}{2\alpha})$ *if* $2\alpha > 1$ *and* $2\beta > 1$ *and otherwise it is* $0$. *Then there exists absolute constants* $M, \tilde{M} > 0$ *such that the following holds*

$$\underline{\mathcal{F}}(\Gamma_2(t)) + \underline{\mathcal{C}} \times \underline{\mathcal{K}}(\Gamma_2(t)) \le \mathcal{P}(t) \le C \times \left(\overline{\mathcal{F}}(\Gamma_2(t)/2) + \overline{\mathcal{C}} \times \overline{\mathcal{K}}(\Gamma_2(t)/2)\right) \quad \text{for all } \Gamma_2(t)B \in [M, d^{2\alpha}\tilde{M}].$$

*where* $\underline{\mathcal{C}}(\gamma_2, B) \overset{def}{=} \int_0^1 \underline{\mathcal{F}}(\Gamma_2(s))\gamma_2^2(s) \ ds$ *and* $C > 0$ *is an absolute constants independent of* $\gamma_2$ *and* $d$.

*On the other hand, if the forcing function with learning rate schedule is non-integrable, that is,*

$$\int_0^\infty h(\Gamma_2(s))\gamma_2^2(s) \ ds = \infty,$$
(89)

*then there exists absolute constant* $M, \tilde{M} > 0$ *such that the following holds*

$$\underline{\mathcal{F}}(\Gamma_2(t)) \le \mathcal{P}(t) \le C \times \overline{\mathcal{F}}(\Gamma_2(t)/2), \quad \text{for all } \Gamma_2(t)B \in [M, d^{2\alpha}\tilde{M}],$$

*where* $C$ *is an absolute constant independent of* $d$, $\gamma_2$, *and* $B$.

*Proof.* Let us first consider the case when (88) holds, that is,

$$\overline{\mathcal{C}} = \int_0^\infty \overline{\mathcal{F}}(\Gamma_2(s))\gamma_2^2(s) \ ds < \infty.$$

Now we consider the upper and lower bound separately.

*Upper bound:* From Proposition F.2, we have that

$$\mathcal{P}(t) \le \overline{\mathcal{F}}(\Gamma_2(t)) + \int_0^t \overline{\mathcal{K}}(\Gamma_2(t) - \Gamma_2(s))\overline{\mathcal{F}}(s)\gamma_2^2(s) \ ds.$$

By a change of variables $(v = \Gamma_2(s))$ and setting $u = \Gamma_2(t)$, we have that the RHS of the above inequality equals

$$\overline{\mathcal{F}}(\Gamma_2(t)) + \int_0^t \overline{\mathcal{K}}(\Gamma_2(t) - \Gamma_2(s))\overline{\mathcal{F}}(s)\gamma_2^2(s) \ ds = \overline{\mathcal{F}}(u) + \int_0^u \overline{\mathcal{K}}(u - v)\overline{\mathcal{F}}(v)g(v) \ dv$$
$$\text{where } g(v) = \gamma_2(\Gamma_2^{-1}(v)).$$

Now let us decompose the convolution into two terms

$$\int_0^u \overline{\mathcal{K}}(u - v)\overline{\mathcal{F}}(v)g(v) \ dv = \int_0^{u/2} \overline{\mathcal{K}}(u - v)\overline{\mathcal{F}}(v)g(v) \ dv + \int_{u/2}^u \overline{\mathcal{K}}(u - v)\overline{\mathcal{F}}(v)g(v) \ dv$$
$$\le \overline{\mathcal{K}}(u/2)\left(\int_0^{u/2} \overline{\mathcal{F}}(v)g(v) \ dv\right) + g(u/2)\overline{\mathcal{F}}(u/2)\int_0^{u/2} \overline{\mathcal{K}}(v) \ dv$$
$$\le \overline{\mathcal{K}}(u/2) \times \left(\int_0^{u/2} \overline{\mathcal{F}}(v)g(v) \ dv\right) + \overline{\gamma}_2\|\overline{\mathcal{K}}\| \times \overline{\mathcal{F}}(u/2).$$

Here we can bound $\overline{\gamma}_2\|\overline{K}\| \le 2$ since we are assuming sufficient conditions on the learning rate for bounded solutions. Now we need to consider the first term. For this, we know by Prop. F.6, Prop. F.3,

Prop. F.4, and Prop. F.5 that there exists $M, \tilde{M} > 0$ such that for all $2Bu \in [M, d^{2\alpha}\tilde{M}]$, we have $\overline{\mathcal{F}}(u/2) \sim h(u/2) + d^{-2\alpha+\max\{1-2\beta,0\}}$ and $\overline{\mathcal{F}}(u/2) \lesssim 1$ for all $2Bu \leq M$. It follows that

$$\int_0^{u/2} \overline{\mathcal{F}}(v)g(v) \; \mathrm{d}v \lesssim \int_0^M \overline{\mathcal{F}}(v)g(v) \; \mathrm{d}v + \int_M^{u/2} h(v)g(v) \; \mathrm{d}v + \int_M^{u/2} d^{-2\alpha+\max\{0,1-2\beta\}} \; \mathrm{d}v.$$
(90)

Here $\lesssim$ means an absolute constant independent of $d$, $\gamma_2$, and $B$. The first and third integrals are bounded by absolute constants and the middle integral is upper bounded by $\overline{\mathcal{C}}$. This proves the upper bound.

_Lower bound:_ From Proposition F.2, we have the following lower bound on the loss curve which after a change of variables gives

$$\underline{\mathcal{F}}(u) + \int_0^u \underline{\mathcal{K}}(u-v)\underline{\mathcal{F}}(v)g(v) \; \mathrm{d}v \leq \mathcal{P}(t).$$

We immediately have the following bound

$$\int_0^u \underline{\mathcal{K}}(u-v)\underline{\mathcal{F}}(v)g(v) \; \mathrm{d}v \geq \underline{\mathcal{K}}(u) \int_0^u \underline{\mathcal{F}}(v)g(v) \; \mathrm{d}v \geq \underline{\mathcal{K}}(u) \int_0^{\Gamma_2(1)} \underline{\mathcal{F}}(v)g(v) \; \mathrm{d}v$$

as $\underline{\mathcal{K}}$ is a decreasing function and $g(v)\underline{F}(v)$ are non-negative.

Next we suppose that

$$\int_0^\infty h(\Gamma_2(s))\gamma_2^2(s) \; \mathrm{d}s = \infty.$$
(91)

_Upper bound:_ To prove the upper bound, we recall that

$$\int_0^u \overline{\mathcal{K}}(u-v)\overline{\mathcal{F}}(v)g(v) \; \mathrm{d}v \leq \overline{\mathcal{K}}(u/2)\left(\int_0^{u/2} g(v)\overline{\mathcal{F}}(v) \; \mathrm{d}v\right) + \overline{\gamma}_2\overline{\mathcal{F}}(u/2)\|\overline{\mathcal{K}}\|.$$

As before, we know by Prop. F.6, Prop. F.3, Prop. F.4, and Prop. F.5 as well as Prop. F.7 and Prop. F.8 that there exists $M, \tilde{M} > 0$ such that for all $2Bu \in [M, d^{2\alpha}\tilde{M}]$, we have $\overline{\mathcal{F}}(u/2) \sim h(u/2) + d^{-2\alpha+\max\{1-2\beta,0\}}$ and $\overline{\mathcal{F}}(u/2) \lesssim 1$ for all $2Bu \leq M$. Moreover $\overline{\mathcal{K}}(u/2) \sim u^{-2+1/(2\alpha)}$ and $\overline{\mathcal{K}}(u/2) \lesssim 1$ for all $2Bu \leq M$.

It follows from (90) that

$$\int_0^{u/2} \overline{\mathcal{F}}(v)g(v) \; \mathrm{d}v \lesssim \int_0^M \overline{\mathcal{F}}(v)g(v) \; \mathrm{d}v + \int_M^{u/2} h(v)g(v) \; \mathrm{d}v + \int_M^{u/2} d^{-2\alpha+\max\{0,1-2\beta\}} \; \mathrm{d}v.$$
(92)

Here $\lesssim$ means an absolute constant independent of $d$, $\gamma_2$, and $B$. We also know that the first and third integrals are bounded by absolute constants. As (91) holds, we know there are two cases to consider.

First suppose $2\alpha < 1$ or $2\beta < 1$ so that $h(t) \sim t^{-(1+\beta/\alpha)+1/(2\alpha)}$. In order for (91) to hold, it must be that $2\beta < 1$. Now we see that if $2\alpha > 1$, then

$$\overline{\mathcal{K}}(u/2) \int_M^{u/2} h(v)g(v) \; \mathrm{d}v \sim u^{-2+1/(2\alpha)}u^{-(1+\beta/\alpha)+1/(2\alpha)+1} \leq u^{-(1+\beta/\alpha)+1/(2\alpha)} \sim \overline{\mathcal{F}}(u/2).$$

Here we use Proposition F.3. Moreover we always have in this regime that $\overline{\mathcal{K}}(u/2) \lesssim \overline{\mathcal{F}}(u/2)$ (here we used Prop. F.3 and Prop. F.7). Thus, all three integrals in (92) are bounded by $\overline{\mathcal{F}}(u/2)$, proving the result in this setting.

Now we suppose that $2\alpha < 1$. As in the prior case $\overline{\mathcal{K}}(u/2) \lesssim \overline{\mathcal{F}}(u/2)$, showing the result for the first and third integrals in (92). It remains to show for the 2nd integral in (92). In this case, we note that the second integral is

$$\overline{\mathcal{K}}(u/2) \int_M^{u/2} h(v)g(v) \; \mathrm{d}v \sim \bar{\gamma}_2 u^{-2+1/(2\alpha)}u^{-(1+\beta/\alpha)+1/(2\alpha)+1}$$

$$\sim d^{2\alpha-1}u^{-2+1/(2\alpha)}u^{-(1+\beta/\alpha)+1/(2\alpha)+1}.$$

Here we used that when $2\alpha < 1$, we need $\bar{\gamma}_2 \lesssim d^{2\alpha-1}$. Now we see that the following hold

$$d^{2\alpha-1}u^{-2+1/(2\alpha)}u^{-(1+\beta/\alpha)+1/(2\alpha)+1} \lesssim u^{-(1+\beta/\alpha)+1/(2\alpha)} \sim \overline{\mathcal{F}}(u/2)$$

$$\Leftrightarrow \quad d^{2\alpha-1}u^{-1+1/(2\alpha)} \lesssim 1.$$

Since $u \in [M, d^{2\alpha}\tilde{M}]$ and $-1 + 1/(2\alpha) > 0$, we see that the left hand side is maximized at $u = d^{2\alpha}\tilde{M}$ which shows that for all $u \in [M, d^{2\alpha}\tilde{M}]$

$$d^{2\alpha-1}u^{-1+1/(2\alpha)} \lesssim d^{2\alpha-1}(d^{2\alpha})^{-1+1/(2\alpha)} \lesssim 1.$$

Hence the result holds in this case.

Now we need to consider the case when $2\alpha > 1$ and $2\beta > 1$. As we have already shown the $\overline{\mathcal{F}}_{pp}(u/2)$ asymptotic is not the reason that (91) holds. Therefore if (91) is true, it must be because the $\overline{\mathcal{F}}_{ac}$ asymptotic is causing the problem. As before, we always have that $\overline{K}(u/2) \lesssim \overline{\mathcal{F}}(u/2)$. Thus, the first and third integral of (92) are done. For the middle integral, we see that

$$d^{-1}u^{-2+1/(2\alpha)}u^{-1+1/(2\alpha)+1} \lesssim u^{-1+1/(2\alpha)} \lesssim \overline{\mathcal{F}}(u/2).$$

This holds precisely because $2\alpha > 1$. Therefore the upper bound in this case is shown.

_Lower bound:_ From Proposition F.2, we have the following lower bound on the loss curve which after a change of variables gives

$$\underline{\mathcal{F}}(u) \le \underline{\mathcal{F}}(u) + \int_0^u \underline{\mathcal{K}}(u-v)\underline{\mathcal{F}}(v)g(v) \; \mathrm{d}v \le \mathcal{P}(t).$$

This proves the result. □

**Remark F.3.** _While we do not have asymptotics for the kernel function when $\alpha < 1/4$, we believe that the kernel function is not power law but rather exponentially decaying. This exponential decay would mean that the forcing function (which is still power law when $\alpha < 1/4$) dominates (above and below) the loss function $\mathcal{P}(t)$._

**Corollary F.2.** _Suppose $\gamma_2(t)$ is a decreasing learning rate schedule (i.e., Assumption 4 holds) and suppose the assumptions of Proposition F.2 hold. Let $\alpha > 1/4$. First suppose that the forcing function with the decaying learning rate is integrable, that is,_

$$\overline{\mathcal{C}} \stackrel{def}{=} \overline{\mathcal{C}}(\gamma_2, B) \stackrel{def}{=} \int_0^\infty h(\Gamma_2(s))\gamma_2^2(s) \; \mathrm{d}s < \infty,$$

_where_ $\quad h(t) \stackrel{def}{=} (2\alpha)^{-1}\Gamma(\frac{\beta}{\alpha} - \frac{1}{2\alpha} + 1) \times (2Bt)^{-(1+\beta/\alpha)+1/(2\alpha)} + c_\beta \times (2Bt)^{-1+1/(2\alpha)} \times d^{-1}.$
(93)

_Here $c_\beta \stackrel{def}{=} \left(\sum_{j=1}^v j^{-2\beta}\right)(2\alpha)^{-1}\Gamma(1 - \frac{1}{2\alpha})$ if $2\alpha > 1$ and $2\beta > 1$ and otherwise it is $0$. Then there exists absolute constants $M, \tilde{M} > 0$ such that the following holds_

$$\underline{\mathcal{F}}(\Gamma_2(t)) + \underline{\mathcal{C}} \times \underline{\mathcal{K}}(\Gamma_2(t)) \le \mathcal{P}(t) \le C \times \left(\overline{\mathcal{F}}(\Gamma_2(t)) + \overline{\mathcal{C}} \times \overline{\mathcal{K}}(\Gamma_2(t))\right) \quad \text{for all } \Gamma_2(t)B \in [M, d^{2\alpha}\tilde{M}].$$

_where $\underline{\mathcal{C}}(\gamma_2, B) \stackrel{def}{=} \int_0^1 \underline{\mathcal{F}}(\Gamma_2(s))\gamma_2^2(s) \; \mathrm{d}s$ and $C > 0$ is an absolute constants independent of $\gamma_2$ and $d$._

_On the other hand, if the forcing function with learning rate schedule is non-integrable, that is,_

$$\int_0^\infty h(\Gamma_2(s))\gamma_2^2(s) \; \mathrm{d}s = \infty,$$
(94)

_then there exists absolute constant $M, \tilde{M} > 0$ such that the following holds_

$$\underline{\mathcal{F}}(\Gamma_2(t)) \le \mathcal{P}(t) \le C \times \overline{\mathcal{F}}(\Gamma_2(t)), \quad \text{for all } \Gamma_2(t)B \in [M, d^{2\alpha}\tilde{M}],$$

_where $C$ is an absolute constant independent of $d$, $\gamma_2$, and $B$._

_Proof._ The result immediately follows from $\overline{\mathcal{F}}(u/2) \sim \underline{\mathcal{F}}(u)$ by Propositions F.4, F.5, F.3, and F.6 and $\overline{\mathcal{K}}(u/2) \sim \underline{\mathcal{K}}(u)$ by Propositions F.7 and F.8 for the kernel function. □

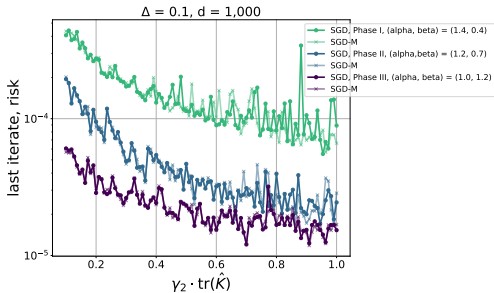

Figure 16: **Comparison between SGD and SGD-M.** Numerical set-up: Run SGD with constant $\gamma_2^{\text{SGD}}$ (14) and SGD-M with $\Delta \equiv 0.1$ and constant $\gamma_3^{\text{SGD-M}}$ (15) on the PLRF model with batch size 1. SGD learning rate $\gamma_2^{\text{SGD}}$ ranges $0.1/\text{Tr}(\hat{K})$ to $1.0/\text{Tr}(\hat{K})$ (100 equally spaced points where selected) where $\text{Tr}(\hat{K}) = \sum_{i=1}^{d} j^{-2\alpha}$. Learning rate $\gamma_3^{\text{SGD-M}} = \Delta \times \gamma_2^{\text{SGD}}$. SGD and SGD-M were run for $10^4$ iterations and the last iterate was recorded. As you can see the solid (dot) lines, SGD, nearly match the same value as the 'x' marked faded lines, SGD-M. This agrees with our theoretical finding that show SGD and SGD-M under the equivalence $\gamma_2^{\text{SGD}} = \frac{\gamma_3^{\text{SGD-M}}}{\Delta}$ have the exact same scaling law in the small batch regime.

## G  Classic Stochastic Momentum (SGD-M)

In this section, we consider the classic (constant) stochastic momentum algorithm (15) under the simplified ODEs (43). In this case, $\delta, \gamma_2, \gamma_3$ are constants independent of $d$ and $\gamma_1 \equiv 1$. In particular, we derive the asymptotics for the forcing function, $\mathcal{F}$, and kernel function $\mathcal{K}$ (see Table 8 for summary). See also Fig. 17 for a summary of the different loss curve behaviors depending on phases in the $(\alpha, \beta)$-plane (Fig 12a for SGD/SGD-M phase diagram).

To begin with, we need to give an expression for the forcing and kernel functions; specifically, we need to solve the ODEs in (43).

### G.1  Solution to the ODE for classic stochastic momentum

Given the Volterra equation (55), we need to understand $\Phi_\lambda(t, s)$ where $\Phi_\lambda(t, s)$ solves the ODE

$$\frac{\mathrm{d}}{\mathrm{d}t}\Phi_\lambda(t, s) = \Omega(t; \lambda)\Phi_\lambda(t, s) \quad \text{such that } \Phi_\lambda(s, s) = \text{Id}_3 \tag{95}$$

where for this constant momentum algorithm, the matrix $\Omega(t, \lambda)$ reduces to

$$\Omega^\lambda \stackrel{\text{def}}{=} \Omega(t; \lambda) = \begin{pmatrix} -2\gamma_2 B\lambda & 0 & -2\gamma_3 \\ 0 & -2\delta & 2B\lambda \\ B\lambda & -\gamma_3 & -\delta - \gamma_2 B\lambda \end{pmatrix}. \tag{96}$$

Since $\Omega^\lambda$ is constant (independent of time), a solution to (43), $\Phi_\lambda(t, s) = \exp(\Omega^\lambda \times (t - s))$, see [25, Chapter 3, Section2] for systems of constant coefficient linear ODEs. As a result, we just need an expression for $\exp(\Omega^x \times t)$.

Computing the exponential of a matrix can easily be done if one knows the eigenvalues and eigenvectors/generalized eigenvectors of the matrix. For example, if the eigenvalues of $\Omega^x$ are distinct or the eigenvectors of $\Omega^x$ form a basis of $\mathbb{R}^3$, then we can write $\exp(\Omega^x \times t) = VD(t)V^T$ where $D(t) = \text{Diag}(\exp(r_i \times t) : i = 1, 2, 3$ and $r_i$ eigenvalue of $\Omega^x)$ and the columns of $V$ are the eigenvectors of $\Omega^x$. In the case that this does not occur, then we can use the Jordan form of $\Omega^x$ to find $\exp(\Omega^x \times t)$. It would involve computing the eigenvalues and generalized eigenvalues of $\Omega^x$.

### G.2  Asymptotics of the kernel and forcing functions, SGD-M

We can now give an explicit solution to the ODE (95) and use it to obtain an explicit formulation for the loss curve $\mathscr{P}$ under the SDE in (40) (see also simplified ODEs (43)).

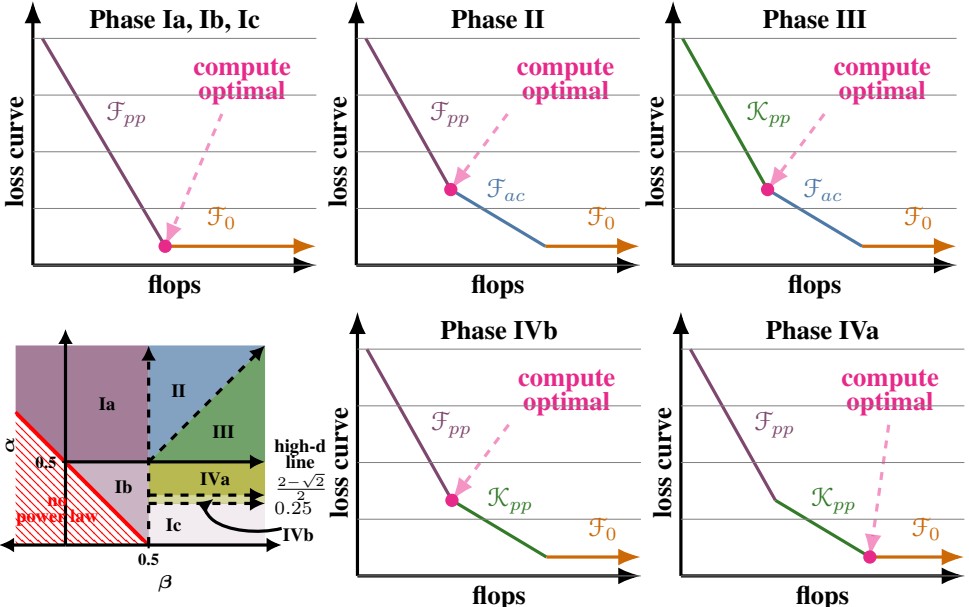

Figure 17: **SGD-M (and SGD) cartoon plots and phase diagram.** The loss curve is the sum of $\mathcal{P}(t) \asymp \mathcal{F}_{pp}(t) + \mathcal{F}_{ac}(t) + \frac{1}{\gamma B}\mathcal{K}_{pp}(t) + \mathcal{F}_0(t)$, where $\gamma \overset{\text{def}}{=} \gamma_2$ for SGD and $\gamma \overset{\text{def}}{=} \gamma_2 + \frac{\gamma_3}{\delta}$ for SGD-M. Each of these terms are explicit and take the form $t^{-\sigma}d^{-\tau}$ (See Table 8 for summary of the asymptotics of the terms and Section G.2 for derivations). Depending on the strength of $\alpha$ relative to $\beta$, some of the terms drop out (Sec. E.1). Compute-optimal derivations can be found in Sec. E.1.

Define $\omega \overset{\text{def}}{=} (\delta - \gamma_2 B\sigma^2)^2 - 4\gamma_3 B\sigma^2$, $\rho \overset{\text{def}}{=} \delta + \gamma_2 B\sigma^2$ $\mu \overset{\text{def}}{=} \delta - \gamma_2 B\sigma^2$.

Then the eigenvalues of $\Omega^\lambda$ are $-\rho, -\rho + \sqrt{\omega}, -\rho - \sqrt{\omega}$ and we write

$$
\begin{aligned}
\Phi_{11}^\lambda(t,s) &= \frac{1}{2\omega}e^{-(t-s)\rho}(-4\gamma_3 B\sigma^2 + e^{-(t-s)\sqrt{\omega}}[(\omega + 2B\sigma^2\gamma_3) - \mu\sqrt{\omega}] \\
&\quad + e^{(t-s)\sqrt{\omega}}[(\omega + 2B\sigma^2\gamma_3) + \mu\sqrt{\omega}]) \\
&= \frac{4B\sigma^2\gamma_3}{2\omega}e^{-(t-s)\rho}(\cosh((t-s)\sqrt{\omega} + \mathcal{V}) - 1)
\end{aligned}
$$

$$
\Phi_{12}^\lambda(t,s) = \frac{2\gamma_3^2}{\omega}e^{-(t-s)\rho}(\cosh((t-s)\sqrt{\omega}) - 1)
$$

Here $\mathcal{V} \in \mathbb{C}$ is defined by $e^{\mathcal{V}} = \sqrt{\frac{(\omega + 2B\sigma^2\gamma_3) + \mu\sqrt{\omega}}{(\omega + 2B\sigma^2\gamma_3) - \mu\sqrt{\omega}}}$.

We can now give precise bounds on $\Phi^\lambda(t,s)_{11}, \Phi^\lambda(t,s)_{12}$.

In the following, we will restrain the study to some values of $\sigma, \delta, \gamma_2, \gamma_3, B$.

**Assumption 5** (Hyperparameter domain). *Suppose that $\delta > 0$, $\gamma_2 > 0$, $\gamma_3 > 0$, $\sigma \in [0,2]$, $B \in \mathbb{N}^*$ and that additionally*

$$
\max\{\gamma_2 B\sigma^2, \frac{\gamma_3}{\delta}B\sigma^2\} \leq \frac{\delta}{4}.
$$

**Proposition G.1.** *There exists $C > 0$ such that under Assumption 5 we have*

$$
\frac{\delta^2}{C} \leq \omega \leq C\delta^2,
$$

$$
\mathcal{V} \in \mathbb{R}_+, \quad \text{and more precisely} \quad \frac{\omega}{C\sigma^2 B\gamma_3} \leq e^{\mathcal{V}} \leq \frac{C\omega}{\sigma^2\gamma_3 B}.
$$

*Proof.* We clearly have $\delta^2 \geq \omega \geq (\frac{3}{4}\delta)^2 - \frac{\delta^2}{4} \geq \frac{\delta^2}{4}$. Additionally, it is clear that $(\omega + 2B\sigma^2\gamma_3) - \mu\sqrt{\omega} > 0$. This implies that $\mathcal{V} \in \mathbb{R}_+$ and more precisely

$$
e^{\mathcal{V}} = \sqrt{\frac{(\omega + 2B\sigma^2\gamma_3) + \mu\sqrt{\omega}}{(\omega + 2B\sigma^2\gamma_3) - \mu\sqrt{\omega}}}
$$

$$
\asymp \sqrt{\frac{\delta^2}{(\delta - \gamma_2\sigma^2)^2 - 2\gamma_3 B\sigma^2 - (\delta - \gamma_2 B\sigma^2)\sqrt{(\delta - \gamma_2 B\sigma^2)^2 - 4\gamma_3 B\sigma^2}}}
$$

$$
\asymp \sqrt{\frac{\delta^2}{(\delta - \gamma_2\sigma^2)^2 - 2\gamma_3 B\sigma^2 - (\delta - \gamma_2 B\sigma^2)^2\sqrt{1 - \frac{4\gamma_3 B\sigma^2}{(\delta - \gamma_2 B\sigma^2)^2}}}}
$$

$$
\asymp \sqrt{\frac{\delta^2}{(\delta - \gamma_2 B\sigma^2)^2\left(\frac{4\gamma_3 B\sigma^2}{(\delta - \gamma_2 B\sigma^2)^2}\right)^2}}
$$

$$
\asymp \sqrt{\frac{\delta^4}{(\gamma_3 B\sigma^2)^2}}
$$

$$
\asymp \frac{\omega}{\gamma_3 B\sigma^2}.
$$

$\square$

**Proposition G.2.** *There exists $C > 0$ such that under Assumption 5 for any $t \geq s \geq 0$ we have*

$$
\frac{1}{C}e^{(t-s)(\sqrt{\omega}-\rho)} \leq \Phi^\lambda(t,s)_{11} \leq Ce^{(t-s)(\sqrt{\omega}-\rho)}
$$

*if additionally $t - s \geq \frac{1}{\delta}$,* $\quad \dfrac{1}{C}\dfrac{\gamma_3^2}{\delta^2}e^{(t-s)(\sqrt{\omega}-\rho)} \leq \Phi^\lambda(t,s)_{12} \leq C\dfrac{\gamma_3^2}{\delta^2}e^{(t-s)(\sqrt{\omega}-\rho)}$

*while if $t - s \leq \frac{1}{\delta}$,* $\quad \dfrac{1}{C}\dfrac{\gamma_3^2}{\delta^2}((t-s)\delta)^2 \leq \Phi^\lambda(t,s)_{12} \leq C\dfrac{\gamma_3^2}{\delta^2}((t-s)\delta)^2$

*In particular one can also choose $C > 0$ such that*

$$
\frac{1}{C}e^{-\frac{1}{C}(t-s)(\gamma_2 + \frac{2\gamma_3}{\delta})B\sigma^2} \leq \Phi^\lambda(t,s)_{11} \leq Ce^{-C(t-s)(\gamma_2 + \frac{2\gamma_3}{\delta})B\sigma^2}
$$

*if additionally $t - s \geq \frac{1}{\delta}$,* $\quad \dfrac{1}{C}\dfrac{\gamma_3^2}{\delta^2}e^{-\frac{1}{C}(t-s)(\gamma_2 + \frac{2\gamma_3}{\delta})B\sigma^2} \leq \Phi^\lambda(t,s)_{12} \leq C\dfrac{\gamma_3^2}{\delta^2}e^{-C(t-s)(\gamma_2 + \frac{2\gamma_3}{\delta})B\sigma^2}$

*Proof.* Since $\mathcal{V} \gtrsim \frac{\omega}{\gamma_3 B\sigma^2} \gtrsim 1$ we know that

$$
\Phi^\lambda_{11}(t,s) \asymp \frac{4B\sigma^2\gamma_3}{2\omega}e^{-(t-s)\rho}(\cosh((t-s)\sqrt{\omega} + \mathcal{V}) - 1)
$$

$$
\asymp \frac{4B\sigma^2\gamma_3}{2\omega}e^{-(t-s)\rho}\exp\left((t-s)\sqrt{\omega} + \mathcal{V}\right)
$$

$$
\asymp \exp((t-s)(\sqrt{\omega} - \rho)).
$$

For the same reason, if $(t-s)\sqrt{\omega} \asymp (t-s)\delta \gtrsim 1$, we have that

$$
\Phi^\lambda(t,s)_{12} \asymp \frac{\gamma_3^2}{\delta^2}e^{(t-s)(\sqrt{\omega}-\rho)}
$$

while if $(t-s)\delta \leq 1$ we use that $\frac{1}{C}((t-s)\delta))^2 \leq \cosh((t-s)\delta) - 1 \leq C((t-s)\delta)^2$

Using Assumption 5 we obtain that there exists $C > 0$ such that

$$\delta + C\left(-\frac{\gamma_2 B\sigma^2}{\delta} + \frac{1}{2}\left(\frac{\gamma_2 B\sigma^2}{\delta}\right)^2 - 2\frac{\gamma_3 B\sigma^2}{\delta^2}\right) \leq \sqrt{\omega}$$

$$= \delta\sqrt{1 - 2\frac{\gamma_2 B\sigma^2}{\delta} + \left(\frac{\gamma_2 B\sigma^2}{\delta}\right)^2 - 4\frac{\gamma_3 B\sigma^2}{\delta^2}}$$

$$\leq \delta + \frac{1}{C}\left(-\frac{\gamma_2 B\sigma^2}{\delta} + \frac{1}{2}\left(\frac{\gamma_2 B\sigma^2}{\delta}\right)^2 + 2\frac{\gamma_3 B\sigma^2}{\delta^2}\right)$$

where we used that $\frac{1}{2} \leq -2\frac{\gamma_2 B\sigma^2}{\delta} + \left(\frac{\gamma_2 B\sigma^2}{\delta}\right)^2 - 4\frac{\gamma_3 B\sigma^2}{\delta^2} \leq 0$ is bounded under Assumption 5.

Additionally, since $\frac{\gamma_2 B\sigma^2}{\delta} \leq \frac{1}{4}$, we can further reduce to

$$\frac{1}{C}(2\gamma_2 + \frac{2\gamma_3}{\delta})B\sigma^2 \leq \sqrt{\omega} - \rho \leq C(2\gamma_2 + \frac{2\gamma_3}{\delta})B\sigma^2.$$

This implies the last result. □

Now that we know the behavior of $\Phi_{11}^\lambda(t, s)$, $\Phi_{12}^\lambda(t, s)$ we can reduce the study to simplified functions that will be used to bound the forcing and kernel functions. This is done in the next definition.

**Definition G.1.** *Let $\mathfrak{C} > 0$ be a large universal constant corresponding to Proposition G.2. Then, under Assumption 5 we define*

$$\begin{cases} \hat{\psi}(\sigma^2, t) \overset{def}{=} e^{-(\frac{\gamma_3}{\delta} + \gamma_2)B\sigma^2 t} \\ \underline{\psi}_{\mathcal{F}}(\sigma^2, t) \overset{def}{=} \frac{1}{\mathfrak{C}}e^{-\mathfrak{C}(\frac{\gamma_3}{\delta} + \gamma_2)B\sigma^2 t} \\ \overline{\psi}_{\mathcal{F}}(\sigma^2, t) \overset{def}{=} \mathfrak{C}e^{-\frac{1}{\mathfrak{C}}(\frac{\gamma_3}{\delta} + \gamma_2)B\sigma^2 t} \\ \underline{\psi}_{\mathcal{K}}(\sigma^2, t) \overset{def}{=} \frac{1}{\mathfrak{C}}\frac{\gamma_3^2}{\delta^2}\left(e^{-\mathfrak{C}(\frac{\gamma_3}{\delta} + \gamma_2)B\sigma^2 t}\mathbb{1}_{t \geq \delta^{-1}} + ((t-s)\delta)^2\mathbb{1}_{t < \delta^{-1}}\right) \\ \overline{\psi}_{\mathcal{K}}(\sigma^2, t) \overset{def}{=} \mathfrak{C}\frac{\gamma_3^2}{\delta^2}\left(e^{-\frac{1}{\mathfrak{C}}(\frac{\gamma_3}{\delta} + \gamma_2)B\sigma^2 t}\mathbb{1}_{t \geq \delta^{-1}} + ((t-s)\delta)^2\mathbb{1}_{t < \delta^{-1}}\right). \end{cases}$$

*Then it is clear from the above that $\forall t \geq s \geq 0$,*

$$\underline{\psi}_{\mathcal{F}}(\sigma^2, t) \leq \Phi^\lambda(t, 0)_{11} \leq \overline{\psi}_{\mathcal{F}}(\sigma^2, t)$$

$$\underline{\psi}_{\mathcal{K}}(\sigma^2, t-s) \leq \Phi^\lambda(t, s)_{12} \leq \overline{\psi}_{\mathcal{K}}(\sigma^2, t-s).$$

*We hence define:*

$$\begin{cases} \tilde{\mathcal{F}}(t) = \int_0^\infty \hat{\psi}(\sigma^2, t)\,\mu_{\mathcal{F}}(d\sigma^2) \\ \underline{\mathcal{F}}(t) = \int_0^\infty \underline{\psi}_{\mathcal{F}}(\sigma^2, t)\,\mu_{\mathcal{F}}(d\sigma^2) \\ \overline{\mathcal{F}}(t) = \int_0^\infty \overline{\psi}_{\mathcal{F}}(\sigma^2, t)\,\mu_{\mathcal{F}}(d\sigma^2) \\ \tilde{\mathcal{K}}_s(t) = B \times \int_0^\infty \sigma^4(\gamma_2^2 + (\frac{\gamma_3}{\delta})^2)\hat{\psi}(\sigma^2, t)\,\mu_{\mathcal{K}}(d\sigma^2) \\ \underline{\mathcal{K}}_s(t) = B \times \int_0^\infty \sigma^4(\gamma_2^2\underline{\psi}_{\mathcal{F}}(\sigma^2, t) + \underline{\psi}_{\mathcal{K}}(\sigma^2, t))\,\mu_{\mathcal{K}}(d\sigma^2) \\ \overline{\mathcal{K}}_s(t) = B \times \int_0^\infty \sigma^4(\gamma_2^2\overline{\psi}_{\mathcal{F}}(\sigma^2, t) + \overline{\psi}_{\mathcal{K}}(\sigma^2, t))\,\mu_{\mathcal{K}}(d\sigma^2). \end{cases}$$

From the above, we see that $\tilde{\mathcal{F}}, \underline{\mathcal{F}}, \overline{\mathcal{F}}$ are non-negative functions and that $\tilde{\mathcal{K}}_s(t) = \tilde{\mathcal{K}}(t-s)$, $\underline{\mathcal{K}}_s(t) = \underline{\mathcal{K}}(t-s)$, $\overline{\mathcal{K}}_s(t) = \overline{\mathcal{K}}(t-s)$ are non-negative convolution kernels. The following proposition shows that to understand the behavior of $\mathcal{P}$, we only need to understand the solutions of the Volterra equations using the bounds on the forcing and kernel functions.

**Proposition G.3.** *We have the bounds for all $0 \leq s \leq t$:*

$$0 \leq \underline{\mathcal{F}}(t) \leq \mathcal{F}(t) \leq \overline{\mathcal{F}}(t),$$

$$0 \leq \underline{\mathcal{K}}_s(t) \leq \mathcal{K}_s(t) \leq \overline{\mathcal{K}}_s(t).$$

In particular, denoting $\mathcal{P}, \underline{\mathcal{P}}, \overline{\mathcal{P}}$ the solutions of the Volterra equations with respectively forcing function $\mathcal{F}, \underline{\mathcal{F}}, \overline{\mathcal{F}}$ and kernel function $\mathcal{K}, \underline{\mathcal{K}}, \overline{\mathcal{K}}$, we have the inequality for all time $t \geq 0$:

$$0 \leq \underline{\mathcal{P}}(t) \leq \mathcal{P}(t) \leq \overline{\mathcal{P}}(t).$$

*Proof.* This is a consequence of proposition G.2 and the fact that $\mu_{\mathcal{F}}, \mu_{\mathcal{K}}$ are positive measures. The last inequality is a consequence of the Volterra equation definition. $\qquad\square$

Denote now, similarly to as was done in Section D:

$$
\begin{aligned}
\tilde{\mathcal{F}}_0(t) &\overset{\text{def}}{=} \mu_{\mathcal{F}}(\{0\})\hat{\psi}(0,t) = \mu_{\mathcal{F}}(\{0\}), \\
\tilde{\mathcal{F}}_{pp}(t) &\overset{\text{def}}{=} \frac{1}{2\alpha} \int_0^1 \sigma^{1+\frac{2\beta-1}{\alpha}} \hat{\psi}(\sigma^2, t)\, \mathrm{d}\sigma, \\
\tilde{\mathcal{F}}_{ac}(t) &\overset{\text{def}}{=} \frac{c_\beta}{2\alpha d} \int_{d^{-\alpha}}^1 \sigma^{1-\frac{1}{\alpha}} \hat{\psi}(\sigma^2, t)\, \mathrm{d}\sigma, \\
\tilde{\mathcal{K}}_{pp}(t) &\overset{\text{def}}{=} \frac{\left(\gamma_2^2 + \left(\frac{\gamma_3}{\delta}\right)^2\right)B}{2\alpha} \int_0^1 \sigma^{3-\frac{1}{\alpha}} \hat{\psi}(\sigma^2, t)\, \mathrm{d}\sigma.
\end{aligned}
\tag{97}
$$

We can compute the following results on $\tilde{\mathcal{F}}_0, \tilde{\mathcal{F}}_{ac}, \tilde{\mathcal{F}}_{pp}, \tilde{\mathcal{K}}_{pp}$ which are summarized in Table 5.

**Proposition G.4.** *Suppose $\alpha, \beta \neq \frac{1}{2}$ and $2\alpha + 2\beta > 1$ and Assumption 5 to hold. For any $M > 0$, then, there exists a constant $C(\alpha, \beta) > 0$ such that we have for all $t \geq 0$:*

$$\frac{1}{C} d^{-2\alpha+\max\{0,1-2\beta\}} \leq \tilde{\mathcal{F}}_0(t) \leq C d^{-2\alpha+\max\{0,1-2\beta\}},$$

$$\frac{1}{C}\min\{1, \left((\gamma_2 + \frac{\gamma_3}{\delta})Bt\right)^{-1-\frac{2\beta-1}{2\alpha}}\} \leq \tilde{\mathcal{F}}_{pp}(t) \leq C\min\{1, \left((\gamma_2 + \frac{\gamma_3}{\delta})Bt\right)^{-1-\frac{2\beta-1}{2\alpha}}\},$$

$$\frac{(\gamma_2^2 + (\frac{\gamma_3}{\delta})^2)B}{C}\min\{1, \left((\gamma_2 + \frac{\gamma_3}{\delta})Bt\right)^{-2+\frac{1}{2\alpha}}\} \leq \tilde{\mathcal{K}}_{pp}(t)$$

$$\leq C(\gamma_2^2 + (\frac{\gamma_3}{\delta})^2)B\min\{1, \left((\gamma_2 + \frac{\gamma_3}{\delta})Bt\right)^{-2+\frac{1}{2\alpha}}\},$$

$$\tilde{\mathcal{F}}_{ac}(t) \leq \begin{cases} C \times \tilde{\mathcal{F}}_0(t) & \text{if } 2\alpha > 1, 2\beta > 1 \\ 0 & \text{if } 2\beta < 1. \end{cases}$$

*If additionally $2\alpha > 1, 2\beta > 1$ and $(\gamma_2 + \frac{\gamma_3}{\delta})B\sigma^2 t \leq Md^{2\alpha}$ we have*

$$\frac{1}{C}d^{-1}\min\{1, \left((\gamma_2 + \frac{\gamma_3}{\delta})Bt\right)^{-1+1/(2\alpha)}\} \leq \tilde{\mathcal{F}}_{ac}(t) \leq Cd^{-1}\left(\min\{1, (\gamma_2 + \frac{\gamma_3}{\delta})Bt\right)^{-1+1/(2\alpha)}.\}$$

*Proof.* The proofs are entirely similar to the one of [80, Proposition H.2, H.3, H.4, H.5] and just follow from the change of variable $u = (\gamma_2 + \frac{\gamma_3}{\delta})B\sigma^2 t$ in the integral definitions (97). For $\tilde{\mathcal{F}}_0$, we used Proposition D.3. $\qquad\square$

**Proposition G.5.** *Suppose $\alpha > 0$, $\alpha, \beta \neq \frac{1}{2}$ and $2\alpha + 2\beta > 1$, $\alpha + 1 > \beta$ and let $M > 0$. Then, there exists a constant $C(\alpha, \beta, M) > 0$ such that we have for all $t \geq s \geq 0$:*

$$\frac{1}{C}\left(\tilde{\mathcal{F}}_0(t) + \tilde{\mathcal{F}}_{ac}(t) + \tilde{\mathcal{F}}_{pp}(t)\right) \leq \tilde{\mathcal{F}}(t) \leq C\left(\tilde{\mathcal{F}}_0(t) + \tilde{\mathcal{F}}_{ac}(t) + \tilde{\mathcal{F}}_{pp}(t)\right).$$

*If additionally $\alpha > \frac{1}{4}$ and $(\gamma_2 + \frac{\gamma_3}{\delta})B(t - s) < Md^{2\alpha}$,*

$$\frac{1}{C}\tilde{\mathcal{K}}_{pp}(t - s) \leq \tilde{\mathcal{K}}(t - s) \leq C\tilde{\mathcal{K}}_{pp}(t - s).$$

*Proof.* The proof is a direct consequence of Propositions D.13 and D.14 and the estimates derived in Section D. We apply Proposition D.13 which implies the existence of $L, C > 0$ such that

$$\frac{1}{C} \int_{Ld^{-2\alpha}}^{\frac{1}{L}} \hat{\psi}(\sigma^2, t)(\mu_{\mathcal{F}_{pp}} + \mu_{\mathcal{F}_{ac}})(\mathrm{d}\sigma^2) \le \int_{Ld^{-2\alpha}}^{\frac{1}{L}} \hat{\psi}(\sigma^2, t)\mu_{\mathcal{F}}(\mathrm{d}\sigma^2)$$

$$\le C \int_{Ld^{-2\alpha}}^{\frac{1}{L}} \hat{\psi}(\sigma^2, t)(\mu_{\mathcal{F}_{pp}} + \mu_{\mathcal{F}_{ac}})(\mathrm{d}\sigma^2).$$

Additionally, we know using Propositions D.4 and D.6 that

$$\max\{\int_0^{cd^{-2\alpha}} \hat{\psi}(\sigma^2, t)(\mu_{\mathcal{F}_{pp}} + \mu_{\mathcal{F}_{ac}})(\mathrm{d}\sigma^2)), \int_0^{cd^{-2\alpha}} \hat{\psi}(\sigma^2, t)\mu_{\mathcal{F}_{pp}}(\mathrm{d}\sigma^2))\} \lesssim \mathcal{F}_0(t).$$

Finally, using Proposition D.2, it is clear that

$$\max\{\int_{1/M}^1 \hat{\psi}(\sigma^2, t)(\mu_{\mathcal{F}_{pp}} + \mu_{\mathcal{F}_{ac}})(\mathrm{d}\sigma^2), \int_{1/M}^1 \hat{\psi}(\sigma^2, t)\mu_{\mathcal{F}}(\mathrm{d}\sigma^2)\} \lesssim \mathcal{F}_{pp}(t).$$

For the kernel, we can similarly write for $L > 0$ large enough

$$\tilde{\mathcal{K}}(t - s) = \frac{(\gamma_2^2 + (\frac{\gamma_3}{\delta})^2)B}{2\alpha} \int_0^1 \hat{\psi}(\sigma^2, t - s)\mu_{\mathcal{K}}(\mathrm{d}\sigma^2)$$

$$= \int_0^{Ld^{-2\alpha}} \hat{\psi}(\sigma^2, t - s)\mu_{\mathcal{K}}(\mathrm{d}\sigma^2) + \int_{Ld^{-2\alpha}}^{1/L} \hat{\psi}(\sigma^2, t - s)\mu_{\mathcal{K}}(\mathrm{d}\sigma^2) + \int_{1/L}^1 \hat{\psi}(\sigma^2, t - s)\mu_{\mathcal{K}}(\mathrm{d}\sigma^2)$$

For the second term, we use Proposition D.14 and that $(\gamma_2 + \frac{\gamma_3}{\delta})B(t - s) < Md^{2\alpha}$ to compare it to the value in Proposition G.4. For the first term, using Proposition D.6 and $\frac{\gamma_3 B}{\delta}(t - s) < Md^{2\alpha}$, it is absorbed by the second term. Finally, the third term is absorbed by the second term due to the exponential decay of $\hat{\psi}$. $\qquad\square$

Now, we can easily bound the forcing and kernel functions using $\tilde{\mathcal{F}}, \tilde{\mathcal{K}}$.

**Proposition G.6.** *Suppose $\alpha, \beta \ne \frac{1}{2}$, $2\alpha + 2\beta > 1$ $\alpha + 1 > \beta$ and let $M > 0$. Then, there exists a constant $C(\alpha, \beta, M) > 0$ such that we have for all $t \ge 0$:*

$$\frac{1}{C}\left(\tilde{\mathcal{F}}_0(t) + \tilde{\mathcal{F}}_{ac}(t) + \tilde{\mathcal{F}}_{pp}(t)\right) \le \underline{\mathcal{F}}(t) \le \overline{\mathcal{F}}(t) \le C\left(\tilde{\mathcal{F}}_0(t) + \tilde{\mathcal{F}}_{ac}(t) + \tilde{\mathcal{F}}_{pp}(t)\right).$$

*If additionally $\alpha > \frac{1}{4}$, $(\gamma_2 + \frac{\gamma_3}{\delta})Bt < Md^{2\alpha}$ and $t \ge \delta^{-1}$,*

$$\frac{1}{C}\tilde{\mathcal{K}}_{pp}(t) \le \underline{\mathcal{K}}(t) \le \overline{\mathcal{K}}(t) \le C\tilde{\mathcal{K}}_{pp}(t).$$

*Finally if $\alpha > \frac{1}{4}$, $(\gamma_2 + \frac{\gamma_3}{\delta})Bt < Md^{2\alpha}$ and $t \le \delta^{-1}$,*

$$\frac{1}{C}\left(\gamma_2^2 B \min\{1, ((\gamma_2 + \frac{\gamma_3}{\delta})Bt)^{-2+1/(2\alpha)}\} + \frac{\gamma_3^2 B}{\delta^2}((t - s)\delta)^2\right) \le \underline{\mathcal{K}}(t) \le \overline{\mathcal{K}}(t)$$

$$\le C\left(\gamma_2^2 B \min\{1, ((\gamma_2 + \frac{\gamma_3}{\delta})Bt)^{-2+1/(2\alpha)}\} + \frac{\gamma_3^2 B}{\delta^2}((t - s)\delta)^2\right).$$

*Proof.* We know that for any $t \ge 0$,

$$\underline{\mathcal{F}}(t) = \frac{1}{\mathfrak{C}}\tilde{\mathcal{F}}(\mathfrak{C}t), \quad \overline{\mathcal{F}}(t) = \mathfrak{C}\tilde{\mathcal{F}}(\frac{1}{\mathfrak{C}}t),$$

$$\text{if additionally } t\delta \ge 1, \quad \underline{\mathcal{K}}(t) = \frac{1}{\mathfrak{C}}\tilde{\mathcal{K}}(\mathfrak{C}t), \quad \overline{\mathcal{K}}(t) = \mathfrak{C}\tilde{\mathcal{K}}(\frac{1}{\mathfrak{C}}t).$$

Additionally, we know that $\left(\tilde{\mathcal{F}}_0(t) + \tilde{\mathcal{F}}_{ac}(t) + \tilde{\mathcal{F}}_{pp}(t)\right)$ and $\tilde{\mathcal{K}}_{pp}(t)$ follow power laws. Hence, we only have to prove the bounds on $\tilde{\mathcal{F}}$ and $\tilde{\mathcal{K}}$ which was done in Proposition G.5. Finally the case $t\delta < 1$ can be handled easily since $\Phi_{12}^\lambda(t, s) \asymp ((t - s)\delta)^2$ and we obtain the result. $\qquad\square$

**Proposition G.7** (Kernel norm bound). *Let $\alpha > \frac{1}{4}$, $\alpha \neq \frac{1}{2}$. There exists some constants $C(\alpha) >$ $0$, $\bar{d}(\alpha) > 1$ such that for $\delta > 0$, $B \in \mathbb{N}^*$, $\gamma_2 > 0$, $\gamma_3 > 0$, if $\gamma_3 B \leq \frac{\delta^2}{16}$, $\gamma_2 B \leq \frac{\delta^2}{16}$, and $d \geq \bar{d}(\alpha)$, we have the bounds (note that for $2\alpha < 1$, we use $v \sim C \times d$)*

$$\frac{d^{(1-2\alpha)_+}}{C}(\gamma_2 + \frac{\gamma_3}{\delta}) \leq \|\underline{\mathcal{K}}\| \leq \|\overline{\mathcal{K}}\|$$

$$\leq Cd^{(1-2\alpha)_+}(\gamma_2 + \frac{\gamma_3}{\delta}).$$

*Proof.* We will instead show

$$\frac{d^{(1-2\alpha)_+}}{C}\frac{\gamma_2^2 + (\frac{\gamma_3}{\delta})^2}{\gamma_2 + \frac{\gamma_3}{\delta}} \leq \|\underline{\mathcal{K}}\| \leq \|\overline{\mathcal{K}}\|$$

$$\leq Cd^{(1-2\alpha)_+}\left(\frac{(\gamma_2^2 + (\frac{\gamma_3}{\delta})^2)B}{(\gamma_2 + \frac{\gamma_3}{\delta})B} + \frac{\gamma_3^2 B}{\delta^3}\right)$$

which directly brings the result as $\forall a, b > 0$ we have $\frac{a^2+b^2}{a+b} \asymp a + b$ and $\frac{\gamma_3^2 B}{\delta^3} \lesssim \frac{\gamma_3}{\delta}$ since $\frac{\gamma_3 B}{\delta^2} \lesssim 1$.

It is clear that $\|\underline{\mathcal{K}}\| \leq \|\overline{\mathcal{K}}\|$. The bounds are then direct consequences from Definition G.1 and the lower and upper-bounds on $\mu_{\mathcal{K}}$ derived in Section D.

We first use Lemma D.1 to get that there exists some $\bar{d} > 1$ large enough such that $\forall d \geq \bar{d}$, $\mu_{\mathcal{K}}([2, \infty)) = 0$. Hence we reduce to $\sigma \leq 2$ and under the assumption of the proposition we see that we are under Assumption 5.

For the lower bound, we write for some $M > 0$ large enough to apply Proposition D.12,

$$\|\underline{\mathcal{K}}\| \gtrsim \int_{t=\delta^{-1}}^{\infty} \int_{\sigma=Md^{-\alpha}}^{\frac{1}{M}} (\gamma_2^2 + (\frac{\gamma_3}{\delta})^2)Be^{-(\gamma_2+\frac{\gamma_3}{\delta})B\sigma^2 t}\,\mathrm{d}\mu_{\mathcal{K}}(\sigma^2)\,\mathrm{d}t$$

$$\gtrsim \int_{t=\delta^{-1}}^{\infty} \int_{\sigma=Md^{-\alpha}}^{\frac{1}{M}} (\gamma_2^2 + (\frac{\gamma_3}{\delta})^2)Be^{-(\gamma_2+\frac{\gamma_3}{\delta})B\sigma^2 t}\,\mathrm{d}\mu_{\mathcal{K}_{pp}}(\sigma^2)\,\mathrm{d}t$$

$$\gtrsim \frac{\gamma_2^2 + (\frac{\gamma_3}{\delta})^2}{\gamma_2 + \frac{\gamma_3}{\delta}} \int_{\sigma=Md^{-\alpha}}^{\frac{1}{M}} \sigma^{3-\frac{1}{\alpha}}\,\mathrm{d}\sigma$$

$$\gtrsim d^{(1-2\alpha)_+}\frac{\gamma_2^2 + (\frac{\gamma_3}{\delta})^2}{\gamma_2 + \frac{\gamma_3}{\delta}}.$$

For the upper-bound, we proceed similarly and write form some $M > 0$ large enough to apply Propositions D.5 and D.9 to D.11

$$\int_{t=\delta^{-1}}^{\infty} \int_{\sigma=0}^{2} (\gamma_2^2 + (\frac{\gamma_3}{\delta})^2)Be^{-(\gamma_2+\frac{\gamma_3}{\delta})B\sigma^2 t}\,\mathrm{d}\mu_{\mathcal{K}}(\sigma^2)\,\mathrm{d}t$$

$$\lesssim \int_{t=\delta^{-1}}^{\infty} \int_{\sigma=\frac{1}{M}d^{-\alpha}}^{2} (\gamma_2^2 + (\frac{\gamma_3}{\delta})^2)Be^{-(\gamma_2+\frac{\gamma_3}{\delta})B\sigma^2 t}\,\mathrm{d}\mu_{\mathcal{K}_{pp}}(\sigma^2)\,\mathrm{d}t$$

$$\lesssim \frac{\gamma_2^2 + (\frac{\gamma_3}{\delta})^2}{\gamma_2 + \frac{\gamma_3}{\delta}} \int_{\sigma=\frac{1}{M}}^{2} \sigma^{3-\frac{1}{\alpha}}\,\mathrm{d}\sigma$$

$$\lesssim d^{(1-2\alpha)_+}\frac{\gamma_2^2 + (\frac{\gamma_3}{\delta})^2}{\gamma_2 + \frac{\gamma_3}{\delta}}.$$

For $\Phi_{11}^{\lambda}(t,s)$, $t - s$ small is not a problem as we have directly

$$\int_{t=0}^{\infty} \int_{\sigma=0}^{2} \gamma_2^2 Be^{-(\gamma_2+\frac{\gamma_3}{\delta})B\sigma^2 t}\,\mathrm{d}\mu_{\mathcal{K}}(\sigma^2)\,\mathrm{d}t$$

$$\lesssim d^{(1-2\alpha)_+}\frac{\gamma_2^2}{\gamma_2 + \frac{\gamma_3}{\delta}}.$$

For $\Phi_{12}^\lambda(t,s)$ we need to be more careful and we write for small times

$$\int_{t=0}^{\delta^{-1}} \int_{\sigma=0}^{2} (\frac{\gamma_3}{\delta})^2 B((t-s)\delta)^2 \, d\mu_{\mathcal{K}}(\sigma^2) \, dt$$

$$\lesssim \int_{t=0}^{\delta^{-1}} \int_{\sigma=\frac{1}{M}d^{-\alpha}}^{2} (\frac{\gamma_3}{\delta})^2 B \, d\mu_{\mathcal{K}_{pp}}(\sigma^2) \, dt$$

$$\lesssim (\frac{\gamma_3}{\delta})^2 B\delta^{-1} \int_{\sigma=\frac{1}{M}}^{2} \sigma^{3-\frac{1}{\alpha}} \, d\sigma$$

$$\lesssim d^{(1-2\alpha)_+} (\frac{\gamma_3}{\delta})^2 B\delta^{-1}.$$

Hence we see that $\|\overline{\mathcal{K}}\| \lesssim d^{(1-2\alpha)_+} \frac{\gamma_2^2+(\frac{\gamma_3}{\delta})^2}{\gamma_2+\frac{\gamma_3}{\delta}} + d^{(1-2\alpha)_+}(\frac{\gamma_3}{\delta})^2 B\delta^{-1}$ which brings the claim and the result.

$\square$

Using the bound Proposition G.7, we obtain the following sufficient condition for the algorithm to be stable.

**Corollary G.1** ((Sufficient) Stability condition for hyperparameters of SGD-M). *Suppose the iterates $\{y_{t-1}, \theta_t\}_{t=0}^{\infty}$ are generated by (15). Let $\delta > 0$, $\gamma_2 > 0$, $\gamma_3 > 0$, $B \in \mathbb{N}^*$. Let $\alpha > \frac{1}{4}$, $\alpha \neq \frac{1}{2}$, $\beta \neq \frac{1}{2}$, $2\alpha + 2\beta > 1$, $\alpha + 1 > \beta$. There exists some constant $c > 0$ such that if $\max\{\gamma_2 B, \frac{\gamma_3 B}{\delta}\} \leq \frac{\delta^2}{16}$ and $\gamma_2 + \frac{\gamma_3}{\delta} \leq cd^{-(1-2\alpha)_+}$, then the solution $\mathcal{P}$ to the simplified Volterra equation Equation (55) remains bounded, i.e. $\|\mathcal{P}\|_\infty < \infty$.*

*Proof.* Notice from Propositions G.5 and G.6 that $\overline{\mathcal{F}}(t)$ is bounded. Additionally, applying Proposition G.7 yields the existence of a corresponding $c > 0$ such that $\|\overline{\mathcal{K}}\| < 1$. We saw in Section C.3 that this implies that the loss remains bounded. $\square$

**Remark G.1** (Necessary stability condition). *Similarly, it is clear using the lower bound on $\|\underline{\mathcal{K}}\|$ from Proposition G.7 that if $\gamma_2 + \frac{\gamma_3}{\delta} \geq Cd^{-(1-2\alpha)_+}$ for some $C(\alpha) > 0$ then the risk is unbounded i.e. $\|\mathcal{P}\|_\infty = \infty$.*

**Proposition G.8.** *Let $\delta > 0$, $\gamma_2 > 0$, $\gamma_3 > 0$, $B \in \mathbb{N}^*$, $M > 0$, $\epsilon > 0$. Let $\alpha > \frac{1}{4}$, $\alpha \neq \frac{1}{2}$. There exists some constant $c > 0$ such that if $\max\{\gamma_2 B, \frac{\gamma_3 B}{\delta}\} \leq \frac{\delta^2}{16}$ and $\gamma_2 + \frac{\gamma_3}{\delta} \leq cd^{-(1-2\alpha)_+}$, then for any $t \geq 0$ with $(\gamma_2 + \frac{\gamma_3}{\delta})Bt \leq Md^{2\alpha}$ we have*

$$\left[\overline{\mathcal{K}} * \overline{\mathcal{K}}\right](t) \leq \epsilon\overline{\mathcal{K}}(t).$$

*Proof.* We apply Propositions G.4 to G.6 to obtain that

$$\text{if } t \geq \delta^{-1}, \quad \overline{\mathcal{K}}(t) \asymp (\gamma_2^2 + (\frac{\gamma_3}{\delta})^2)B\min\{1, ((\gamma_2 + \frac{\gamma_3}{\delta})Bt)^{-2+1/(2\alpha)}\},$$

$$\text{if } t < \delta^{-1}, \quad \overline{\mathcal{K}}(t) \asymp \gamma_2^2 B\min\{1, ((\gamma_2 + \frac{\gamma_3}{\delta})Bt)^{-2+1/(2\alpha)}\} + (\frac{\gamma_3}{\delta})^2 B((t-s)\delta)^2.$$

A crucial observation is that $\overline{\mathcal{K}}$ behaves as a power law, i.e. there exists some constant $C > 0$ such that $\forall t \geq 0$, $\frac{1}{C} \leq \frac{\overline{\mathcal{K}}(2t)}{\overline{\mathcal{K}}(t)} \leq C$. Indeed, if $t < \frac{\delta^{-1}}{2}$ or $t \geq \delta^{-1}$ it is clear. This is still true for $t \in [\frac{\delta^{-1}}{2}, \delta^{-1}]$ by just noticing $\lim_{t\uparrow\delta^{-1}} \overline{\mathcal{K}}(t) \asymp \overline{\mathcal{K}}(\delta^{-1})$ since $(\gamma_2 + \frac{\gamma_3}{\delta})B \lesssim \delta$.

It is additionally clear from Proposition G.7 that for any $\epsilon > 0$, for $c > 0$ small enough we have $\|\overline{\mathcal{K}}\| < \epsilon$.

Using these two previous facts, one just writes for any $t \geq 0$

$$[\overline{\mathcal{K}} * \overline{\mathcal{K}}](t) = \int_0^t \overline{\mathcal{K}}(t-s)\overline{\mathcal{K}}(s)\,\mathrm{d}s$$

$$= \int_0^{t/2} \overline{\mathcal{K}}(t-s)\overline{\mathcal{K}}(s)\,\mathrm{d}s + \int_{t/2}^t \overline{\mathcal{K}}(t-s)\overline{\mathcal{K}}(s)\,\mathrm{d}s$$

$$\lesssim \overline{\mathcal{K}}(t)\int_0^\infty \overline{\mathcal{K}}(s)\,\mathrm{d}s$$

$$\lesssim \epsilon\overline{\mathcal{K}}(t).$$

By decreasing $\epsilon$ as needed we obtain the result. $\qquad\square$

We finally state a result on the forcing function norm (see related result [80, Proposition H.6]).

**Proposition G.9.** *Let $\alpha > 0$, $\alpha \neq \frac{1}{2}$, $\beta \neq \frac{1}{2}$, $\alpha + 1 > \beta$, $2\alpha + 2\beta > 1$. There exists $c > 0$ such that if $\max\{\gamma_2 B, \frac{\gamma_3 B}{\delta}\} \leq \frac{\delta^2}{16}$ and $\gamma_2 + \frac{\gamma_3}{\delta} \leq cd^{-(1-2\alpha)_+}$, then for any $t \geq 0$ with $1 \leq (\gamma_2 + \frac{\gamma_3}{\delta})Bt \leq Md^{2\alpha}$ we have*

*If $2\beta > 1$, $\int_0^t \tilde{\mathcal{F}}(s)\,\mathrm{d}s \asymp \frac{1}{(\gamma_2 + \frac{\gamma_3}{\delta})B}$.*

*If $2\beta < 1$, $\frac{1}{(\gamma_2 + \frac{\gamma_3}{\delta})B}\tilde{\mathcal{K}}_{pp}(t) \lesssim \tilde{\mathcal{K}}_{pp}(t) \times \int_0^t \tilde{\mathcal{F}}(s)\,\mathrm{d}s \lesssim \tilde{\mathcal{F}}(t) + \frac{1}{(\gamma_2 + \frac{\gamma_3}{\delta})B}\tilde{\mathcal{K}}_{pp}(t)$.*

*Proof. First case: $2\beta > 1$*

First the contribution of $\hat{\mathcal{F}}_0$ gives $\int_0^{Md^{2\alpha}/(\gamma_2 + \frac{\gamma_3}{\delta})B} \tilde{\mathcal{F}}_0(s)\,\mathrm{d}s \lesssim \frac{1}{(\gamma_2 + \frac{\gamma_3}{\delta})B}$ because $\tilde{\mathcal{F}}_O(t) \asymp d^{-2\alpha}$.

It is clear that $\tilde{\mathcal{F}}_{pp} \in \mathbb{L}^1(\mathbb{R}_+)$. Hence since $1 \leq (\gamma_2 + \frac{\gamma_3}{\delta})Bt$ we know that $\int_0^t \tilde{\mathcal{F}}_{pp}(s)\,\mathrm{d}s \asymp \frac{1}{(\gamma_2 + \frac{\gamma_3}{\delta})B}$.

For $\tilde{\mathcal{F}}_{ac}$, if $2\alpha < 1$, we handle it like $\tilde{\mathcal{F}}_0$. If on the other hand $2\alpha > 1$ then we write

$$\int_{\frac{1}{(\gamma_2 + \frac{\gamma_3}{\delta})B}}^{Md^{2\alpha}/((\gamma_2 + \frac{\gamma_3}{\delta})B)} \tilde{\mathcal{F}}_{ac}(s)\,\mathrm{d}s \lesssim \frac{d^{-1}}{(\gamma_2 + \frac{\gamma_3}{\delta})B}\int_1^{Md^{2\alpha}} x^{-1 + \frac{1}{2\alpha}}\,\mathrm{d}x$$

$$\lesssim \frac{1}{(\gamma_2 + \frac{\gamma_3}{\delta})B}.$$

Since for $(\gamma_2 + \frac{\gamma_3}{\delta})Bt \leq 1$ we know that $\tilde{\mathcal{F}}_{ac}(t) \lesssim d^{-1}$ we obtain that $\int_0^{Md^{2\alpha}/((\gamma_2 + \frac{\gamma_3}{\delta})B)} \tilde{\mathcal{F}}_{ac}(s)\,\mathrm{d}s \lesssim \frac{1}{(\gamma_2 + \frac{\gamma_3}{\delta})B}$. This concludes for the case $2\beta > 1$.

*2nd case: $2\beta < 1$* In this phase, we do not need to worry about $\tilde{\mathcal{F}}_{ac}$ since it is zero.

It is also clear that $\frac{1}{(\gamma_2 + \frac{\gamma_3}{\delta})B}\tilde{\mathcal{K}}_{pp}(t) \lesssim \tilde{\mathcal{K}}_{pp}(t) \times \int_0^t \tilde{\mathcal{F}}(s)\,\mathrm{d}s$ since $\int_0^t \tilde{\mathcal{F}}(s)\,\mathrm{d}s \gtrsim \int_0^{\frac{1}{(\gamma_2 + \frac{\gamma_3}{\delta})B}} \tilde{\mathcal{F}}_{pp}(s)\,\mathrm{d}s \gtrsim \frac{1}{(\gamma_2 + \frac{\gamma_3}{\delta})B}$.

We first consider the $\tilde{\mathcal{F}}_{pp}$ contribution.

$$\tilde{\mathcal{K}}_{pp}(t) \times \int_0^t \tilde{\mathcal{F}}_{pp}(s)\,\mathrm{d}s \lesssim \tilde{\mathcal{K}}_{pp}(t) \times \left(\int_0^{\frac{1}{\gamma_{\mathrm{eff}}B}} \tilde{\mathcal{F}}_{pp}(s)\,\mathrm{d}s + \int_{\frac{1}{\gamma_{\mathrm{eff}}B}}^t \tilde{\mathcal{F}}_{pp}(s)\,\mathrm{d}s\right)$$

$$\lesssim \frac{1}{\gamma_{\mathrm{eff}}B}\tilde{\mathcal{K}}_{pp}(t) + \tilde{\mathcal{K}}_{pp}(t)\int_{\frac{1}{\gamma_{\mathrm{eff}}B}}^t \tilde{\mathcal{F}}_{pp}(s)\,\mathrm{d}s$$

$$\lesssim \frac{1}{\gamma_{\mathrm{eff}}B}\tilde{\mathcal{K}}_{pp}(t) + \gamma_{\mathrm{eff}}^2 B(\gamma_{\mathrm{eff}}Bt)^{-2 + \frac{1}{2\alpha}}\int_{\frac{1}{\gamma_{\mathrm{eff}}B}}^t (\gamma_{\mathrm{eff}}Bs)^{-1 - \frac{2\beta - 1}{2\alpha}}\,\mathrm{d}s$$

$$\lesssim \frac{1}{\gamma_{\mathrm{eff}}B}\tilde{\mathcal{K}}_{pp}(t) + \gamma_{\mathrm{eff}}(\gamma_{\mathrm{eff}}Bt)^{-2 + \frac{1}{2\alpha}}(\gamma_{\mathrm{eff}}Bt)^{-\frac{2\beta - 1}{2\alpha}}.$$

We only need to show that

$$\gamma_{\text{eff}}(\gamma_{\text{eff}}Bt)^{-2+\frac{1}{2\alpha}}(\gamma_{\text{eff}}Bt)^{-\frac{2\beta-1}{2\alpha}} \lesssim \tilde{\mathcal{F}}_{pp}(t)$$
$$\iff \gamma_{\text{eff}}(\gamma_{\text{eff}}Bt)^{-2+\frac{1}{2\alpha}}(\gamma_{\text{eff}}Bt)^{-\frac{2\beta-1}{2\alpha}} \lesssim (\gamma_{\text{eff}}Bt)^{-1-\frac{2\beta-1}{2\alpha}}$$
$$\lesssim \gamma_{\text{eff}} \lesssim (\gamma_{\text{eff}}Bt)^{1-\frac{1}{2\alpha}}.$$

Since by assumption $\gamma_{\text{eff}}Bt \leq Md^{2\alpha}$ we see that we only need $\gamma_{\text{eff}} \lesssim d^{2\alpha-1}$. This is true since by assumption $\gamma_{\text{eff}} \lesssim d^{-(1-2\alpha)_+} \lesssim d^{2\alpha-1}$.

We now consider the $\tilde{\mathcal{F}}_0$ contribution that we will bound using $\tilde{\mathcal{F}}_{pp}(t)$.

$$\tilde{\mathcal{K}}_{pp}(t) \times \int_0^t \tilde{\mathcal{F}}_0(s)\,\mathrm{d}s \lesssim \tilde{\mathcal{K}}_{pp}(t)d^{-2\alpha-2\beta+1}t$$
$$\lesssim \gamma_{\text{eff}}^2 B(\gamma_{\text{eff}}Bt)^{-2+\frac{1}{2\alpha}}d^{-2\alpha-2\beta+1}t$$
$$\lesssim \gamma_{\text{eff}}(\gamma_{\text{eff}}Bt)^{-1+\frac{1}{2\alpha}}d^{-2\alpha-2\beta+1}.$$

We only need to show

$$\tilde{\mathcal{K}}_{pp}(t) \times \int_0^t \tilde{\mathcal{F}}_0(s)\,\mathrm{d}s \lesssim \tilde{\mathcal{F}}(t)$$
$$\iff \gamma_{\text{eff}}(\gamma_{\text{eff}}Bt)^{-1+\frac{1}{2\alpha}}d^{-2\alpha-2\beta+1} \lesssim (\gamma_{\text{eff}}Bt)^{-1-\frac{2\beta-1}{2\alpha}}$$
$$\iff \gamma_{\text{eff}} \lesssim (\gamma_{\text{eff}}Bt)^{-\frac{\beta}{\alpha}}d^{2\alpha+2\beta-1}$$

Since $\gamma_{\text{eff}}Bt \geq 1$ we only need to show that $\gamma_{\text{eff}} \lesssim d^{2\alpha-1+2\beta}$ which is true since $\gamma_{\text{eff}} \lesssim d^{-(1-2\alpha)_+}$ by assumption and $(1-2\alpha)_+ \geq (1-2\alpha) - 2\beta$.

$\square$

**Theorem G.1.** *Suppose* $\alpha > \frac{1}{4}$, $2\alpha + 2\beta > 1$, $\alpha, \beta \neq \frac{1}{2}$, $\alpha + 1 > \beta$. *Let* $M > 0, \delta > 0, \gamma_2 > 0$, $\gamma_3 > 0$, $B \in \mathbb{N}^*$. *There exists some constant* $c > 0$ *such that if* $\max\{\gamma_2 B, \frac{\gamma_3 B}{\delta}\} \leq \frac{\delta^2}{16}$ *and* $\gamma_2 + \frac{\gamma_3}{\delta} \leq cd^{-(1-2\alpha)_+}$, *then for any* $t \geq 0$ *with* $1 \leq (\gamma_2 + \frac{\gamma_3}{\delta})Bt \leq Md^{2\alpha}$ *and* $t \geq \delta^{-1}$, *we have*

$$c(\tilde{\mathcal{F}}_0(t)+\tilde{\mathcal{F}}_{ac}(t)+\tilde{\mathcal{F}}_{pp}(t)+\frac{1}{(\gamma_2 + \frac{\gamma_3}{\delta})B}\tilde{\mathcal{K}}_{pp}(t)) \leq \mathcal{P}(t) \leq \frac{1}{c}(\tilde{\mathcal{F}}_0(t)+\tilde{\mathcal{F}}_{ac}(t)+\tilde{\mathcal{F}}_{pp}(t)+\frac{1}{(\gamma_2 + \frac{\gamma_3}{\delta})B}\tilde{\mathcal{K}}_{pp}(t)).$$

*Proof.* Using Proposition G.3, it suffices to prove the result on $\underline{\mathcal{P}}(t)$ and $\overline{\mathcal{P}}(t)$. We already know that $\forall t \geq 0$,

$$\underline{\mathcal{F}}(t) + [\underline{\mathcal{F}} * \underline{\mathcal{K}}](t) \leq \underline{\mathcal{P}}(t) \leq \overline{\mathcal{P}}(t) \leq \overline{\mathcal{F}}(t) + \sum_{k=1}^{\infty}\left[\overline{\mathcal{F}} * \overline{\mathcal{K}}^{*k}\right](t).$$

Let $\epsilon = \frac{1}{2}$. Using Proposition G.8 we know that for $c > 0$ small enough, $\forall t \geq 0$ if $(\gamma_2 + \frac{\gamma_3}{\delta})Bt \leq Md^{2\alpha}$,

$$\left[\overline{\mathcal{K}} * \overline{\mathcal{K}}\right](t) \leq \epsilon\overline{\mathcal{K}} \quad \text{and} \quad \|\overline{\mathcal{K}}\| < 1.$$

Hence, applying Lemma C.3 we can bound for some $C > 0$

$$\sum_{k=1}^{\infty}\left[\overline{\mathcal{F}} * \overline{\mathcal{K}}^{*k}\right](t) \leq C \times \left[\overline{\mathcal{F}} * \overline{\mathcal{K}}\right](t).$$

We only have left to respectively lower and upper bound $[\underline{\mathcal{F}} * \underline{\mathcal{K}}](t)$ and $\left[\overline{\mathcal{F}} * \overline{\mathcal{K}}\right](t)$.

We write using Proposition G.6, and for $t \geq \delta^{-1}$ and $(\gamma_2 + \frac{\gamma_3}{\delta})B \geq 1$,

$$\left[\overline{\mathcal{F}} * \overline{\mathcal{K}}\right](t) = \int_0^{t/2} \overline{\mathcal{F}}(s)\overline{\mathcal{K}}(t,s)\,\mathrm{d}s + \int_{t/2}^t \overline{\mathcal{F}}(s)\overline{\mathcal{K}}(t,s)\,\mathrm{d}s$$

$$\lesssim \tilde{\mathcal{K}}_{pp}(t) \times \int_0^{t/2} \overline{\mathcal{F}}(s)\,\mathrm{d}s + (\tilde{\mathcal{F}}_0(t) + \tilde{\mathcal{F}}_{ac}(t) + \tilde{\mathcal{F}}_{pp}(t)) \times \|\overline{\mathcal{K}}\|$$

$$\lesssim \tilde{\mathcal{F}}_0(t) + \tilde{\mathcal{F}}_{ac}(t) + \tilde{\mathcal{F}}_{pp}(t) + \frac{1}{(\gamma_2 + \frac{\gamma_3}{\delta})B} \times \tilde{\mathcal{K}}_{pp}(t).$$

Similarly for the lower-bound we write

$$\left[\underline{\mathcal{F}} * \underline{\mathcal{K}}\right](t) \gtrsim \int_0^{t/2} \underline{\mathcal{F}}(s)\underline{\mathcal{K}}(t,s)\,\mathrm{d}s$$

$$\gtrsim \tilde{\mathcal{K}}_{pp}(t) \times \int_0^{t/2} \underline{\mathcal{F}}(s)\,\mathrm{d}s$$

$$\gtrsim \frac{1}{(\gamma_2 + \frac{\gamma_3}{\delta})B}\tilde{\mathcal{K}}_{pp}(t).$$

Here we used from Proposition G.9 that if $1 \leq (\gamma_2 + \frac{\gamma_3}{\delta})Bt \leq Md^{2\alpha}$, then $\tilde{\mathcal{F}}(t) + \tilde{\mathcal{K}}_{pp}(t)\int_0^t \tilde{\mathcal{F}}(s) \asymp \tilde{\mathcal{F}}(t) + \frac{1}{(\gamma_2 + \frac{\gamma_3}{\delta})B}\tilde{\mathcal{K}}_{pp}(t)$.

Finally, using again Proposition G.5, we obtain the result. $\qquad\square$

**Remark G.2.** *Theorem G.1 together with the asymptotics in Proposition G.4 shows that SGD-M with learning rates $\gamma_1 = 1$, $\gamma_2$, $\gamma_3$ and momentum parameter $\delta$ has the exact same scaling laws than SGD where $\gamma_2 \leftarrow (\gamma_2 + \frac{\gamma_3}{\delta})$.*

# H   DANA-constant

In this section, we analyze the DANA-constant algorithm introduced in Section B.3. We are interested in deriving the forcing and kernel function as well as their asymptotics (See Table 12 for summary). As in SGD-M/SGD, depending on $(\alpha, \beta)$-plane, the loss simplifies. We summarize the results in Figure 18 which shows the different phases and a qualitative description of the loss curves.

We begin with a description of change of variables in the hyperparameters and then go into details about solving the simplied ODEs in (43) with hyperparameters associated with DANA-constant. This ultimately leads to the asymptotic description of the forcing and kernel functions.

First, since $\gamma_1(t), \gamma_3(t)$ are constants, DANA-constant with hyperparameters $(\gamma_1, \gamma_2, \gamma_3, \frac{\delta}{1+t})$ yields the same algorithm as DANA-constant with parameters $(\tilde{\gamma}_1, \gamma_2, \tilde{\gamma}_3, \frac{\delta}{1+t})$ where we chose $\tilde{\gamma}_1 \overset{\text{def}}{=} 1$, $\tilde{\gamma}_3 \overset{\text{def}}{=} \gamma_3 \times \gamma_1$. Indeed, one can check that the updates of the two algorithms on the $\theta$ variable are identical, and also that the forcing and kernel functions are identical. Hence across this section, we will freely use $\gamma_1 = 1$ without loss of generality. Additionally, throughout this section, we use $\sigma^2$ to denote the eigenvalue $\lambda$ of $\hat{K}$, that is $\lambda \overset{\text{def}}{=} \sigma^2$.

Below we introduce the main parametrization for DANA-constant that will often be used throughout this section. It essentially amounts to consider (DANA) with $\kappa_3 = 0$.

**Parametrization H.1.** *Let a vector of hyperparameters $H \overset{\text{def}}{=} (\tilde{\gamma}_2, \tilde{\gamma}_3, c_b, \kappa_1, \kappa_2, \kappa_b, \delta)$ with $\tilde{\gamma}_2, \tilde{\gamma}_3, c_b > 0$, $\kappa_1$, $\kappa_2$, $\kappa_b \geq 0$. We add the restriction $\kappa_b \leq \min\{\kappa_1, \kappa_2\}$, $-2\alpha \leq -\kappa_2 + 2\kappa_1 - \kappa_b \leq 0$. We parametrize*

$$\gamma_1 = 1, \quad \gamma_2 = \tilde{\gamma}_2 d^{-\kappa_1}, \quad \gamma_3 = \tilde{\gamma}_3 d^{-\kappa_2}, \quad B = c_b d^{\kappa_b}. \tag{98}$$

**Remark H.1.** *The reason to require $\kappa_b \leq \min\{\kappa_1, \kappa_2\}$ and $\kappa_1, \kappa_2 \geq 0$ is to ensure that $\gamma_2 B, \gamma_3 B$ remain bounded as $d \to \infty$. Otherwise, eigenvalues of (23) do generally no longer have negative real*

*part and the algorithm would trivially diverge Additionnally we require $-2\alpha \leq -\kappa_2 + 2\kappa_1 - \kappa_b \leq 0$ to ensure $d^{-2\alpha} \lesssim \frac{\gamma_3}{\gamma_2^2 B} \lesssim 1$. This condition is in fact not very restrictive on most scalings of interests. We mostly make it to ensure that no edge case for particularly small schedules or very large batch creates problems in the scaling laws. In particular it is satisfied for any $B \lesssim d$, $\gamma_3 \asymp \gamma_2 \frac{B}{d}$, $d^{-1} \lesssim \gamma_2 \lesssim d^{-(1-2\alpha)_+}$ which includes DANA-constant with batch $B = 1$ in Section 3. In particular note that Parametrization H.1 allows for $B \asymp d$ which the reader can check will allow for outscaling when $2\alpha < 1$ when reported in Theorem 4.1.*

## H.1 Simplification of the ODE

In theory, the Frobenius method to get asymptotic solutions of an ODE can be applied to Equation (22) (see [25, Chapters 4, 5] (with some care near zero for the $\left(\frac{1}{1+t}\right)^2$ term). However, the computations are cumbersome. Instead we will work with the simplified ODE

$$\frac{d\Phi_{\sigma^2}(t)}{dt} = \left( \frac{\begin{pmatrix} 0 & 0 & 0 \\ 0 & -2\delta & 0 \\ 0 & 0 & -\delta \end{pmatrix}}{1+t} + \begin{pmatrix} -2\gamma_2 B\sigma^2 & 0 & -2\gamma_3(t) \\ 0 & 0 & 2\gamma_1 B\sigma^2 \\ \gamma_1 B\sigma^2 & -\gamma_3(t) & -\gamma_2 B\sigma^2 \end{pmatrix} \right) \Phi_{\sigma^2}(t). \quad (99)$$

To simplify the computations, we will additionally do the change of variable $\tilde{\Phi}(t) \overset{\text{def}}{=} \begin{pmatrix} \Phi_1(t) \\ \frac{\gamma_3}{\gamma_1 B\sigma^2}\Phi_2(t) \\ \frac{\sqrt{\gamma_3}}{\sqrt{\gamma_1}B\sigma}\Phi_3(t) \end{pmatrix}$

on Equation (99) and get the new ODE

$$\frac{d\tilde{\Phi}(t)}{dt} = \left( \frac{\begin{pmatrix} 0 & 0 & 0 \\ 0 & -2\delta & 0 \\ 0 & 0 & -\delta \end{pmatrix}}{1+t} + \begin{pmatrix} -2\gamma_2 B\sigma^2 & 0 & -2\sqrt{\gamma_3\gamma_1}B\sigma \\ 0 & 0 & 2\sqrt{\gamma_1\gamma_3}B\sigma \\ \sqrt{\gamma_1\gamma_3}B\sigma & -\sqrt{\gamma_3\gamma_1}B\sigma & -\gamma_2 B\sigma^2 \end{pmatrix} \right) \tilde{\Phi}(t). \quad (100)$$

The following technical lemma shows the decreasing behavior of the norm of the solutions of ODE (100).

**Lemma H.1.** *Consider the ODE (100) on $\Phi \overset{\text{def}}{=} (X, Y, Z) : (-1, \infty) \to \mathbb{R}^3$. Then we have the identity for any $t > -1$*

$$\frac{d(X^2(t) + Y^2(t) + 2Z^2(t))}{dt} = -4\gamma_2 B\sigma^2 X^2(t) - 4\frac{\delta}{1+t}Y^2(t) - 4\left(\frac{\delta}{1+t} + \gamma_2 B\sigma^2\right) Z(t)^2.$$

*Proof.* This comes from the fact that

$$\begin{cases} \frac{dX^2(t)}{dt} = -4\gamma_2 B\sigma^2 X^2(t) - 4\sqrt{\gamma_3\gamma_1}B\sigma X(t)Z(t), \\ \frac{dY^2(t)}{dt} = -4\frac{\delta}{1+t}Y^2(t) + 4\sqrt{\gamma_3\gamma_1}B\sigma Y(t)Z(t), \\ \frac{d2Z^2(t)}{dt} = -4\frac{\delta}{1+t}Z^2(t) + 4\sqrt{\gamma_3\gamma_1}B\sigma X(t)Z(t) - 4\sqrt{\gamma_3\gamma_1}B\sigma Y(t)Z(t) - 4\gamma_2 B\sigma^2 Z^2(t). \end{cases}$$

$\square$

## H.2 Getting asymptotic solutions of the ODE through Frobenius method

In the following, we state two strenghtened results from [25, Chap. 4, Thm 4.1; Chap. 5, Thms 2.1, 4.1] to get *uniform estimates in parameter space* of asymptotic solutions of (100) for small and large $t$.

**Theorem H.1** (Singularity of the 1st kind around 0). *Let $Z \subset \mathbb{R}^n$ for some $n \geq 1$ and define for any $\zeta \in Z$ the ODE: $\frac{d\Phi^\zeta}{dt}(t) = \left(\frac{R^\zeta}{t} + A^\zeta\right)\Phi^\zeta(t)$ where $\Phi^\zeta : \mathbb{R}_+^* \mapsto \mathcal{M}_{3,3}(\mathbb{R})$ and $R^\zeta, A^\zeta \in \mathcal{M}_{3,3}(\mathbb{R})$. Suppose the existence of $\delta > 0, C > 0$ such that $\forall \zeta \in Z$, $R^\zeta = \mathrm{Diag}(r_1^\zeta, r_2^\zeta, r_3^\zeta)$ is diagonal with $\min_{1 \leq i \neq j \leq 3} d(|r_i^\zeta - r_j^\zeta|, \mathbb{N}) \geq \delta$ and that $\|R^\zeta\|_\infty, \|A^\zeta\|_\infty \leq C$. Then we have the following:*

1. *$\forall \zeta \in Z$, $\hat{\Phi}^\zeta(t) = \left(I_3 + \sum_{k \geq 1} P_k^\zeta t^k\right) t^{R^\zeta}$ is a fundamental solution where noting $P_0 = I_3$, the matrices $P_k$ are uniquely defined by the recurrence relation $\forall k \geq 0, P_{k+1}^\zeta \left[R^\zeta + (k+1)I_3\right] = R^\zeta P_{k+1}^\zeta + A^\zeta P_k^\zeta$. Especially, $\forall k \geq 0, \exists C_k, \forall \zeta \in Z, \|P_k^\zeta\| \leq C_k$.*

2. *As a consequence, $\forall K \geq 0, \exists T(\delta, C, K) > 0, \exists D(\delta, C, K) > 0$ such that $\forall \zeta \in Z$, the fundamental solution $\hat{\Phi}^\zeta$ verifies $\forall t \leq T, \forall i \in [3], \|\hat{\Phi}_{:,i}^\zeta - \hat{\Phi}_{:,i}^{\zeta,K}\| \leq Dt^{Re(R_{ii}^\zeta + K+1)}$ where $\hat{\Phi}^{\zeta,K} \stackrel{def}{=} \left(I_3 + \sum_{k \leq K} P_k^\zeta t^k\right) t^{R^\zeta}$.*

**Theorem H.2** (Singularity of the 2nd kind around infinity). *Let $Z \subset \mathbb{R}^n$ for some $n \geq 1$ and define for any $\zeta \in Z$ the ODE: $\frac{d\Phi^\zeta}{dt}(t) = \left(\frac{R^\zeta}{t} + A^\zeta\right)\Phi^\zeta(t)$ where $\Phi^\zeta : \mathbb{R}_+^* \mapsto \mathcal{M}_{3,3}(\mathbb{R})$ and $R^\zeta, A^\zeta \in \mathcal{M}_{3,3}(\mathbb{R})$. Suppose the existence of $\delta > 0, C > 0$ such that $\forall \zeta \in Z$, $A^\zeta = \mathrm{Diag}(\mu_1^\zeta, \mu_2^\zeta, \mu_3^\zeta)$ is diagonal, $\min_{1 \leq i \neq j \leq 3} |\mu_i^\zeta - \mu_j^\zeta| \geq \delta$ and that $\|R^\zeta\|_\infty, \|A^\zeta\|_\infty \leq C$. Then we have the following:*

1. *$\forall \zeta \in Z$, $\hat{\Phi}^\zeta(t) = \left(I_3 + \sum_{k \geq 1} \frac{P_k^\zeta}{t^k}\right) t^{\tilde{R}^\zeta} e^{tA^\zeta}$ is a formal fundamental solution where $\tilde{R}_{ij}^\zeta = \delta_{ij} R_{ij}^\zeta$ and noting $P_0 = I_3$, the matrices $P_k$ are uniquely defined by the recurrence relation $\forall k \geq 0, P_{k+1}^\zeta A^\zeta - A^\zeta P_{k+1}^\zeta = R^\zeta P_k^\zeta - P_k^\zeta \tilde{R}^\zeta + kP_k^\zeta$. Especially, $\forall k \geq 0, \exists C_k, \forall \zeta \in Z, \|P_k^\zeta\| \leq C_k$.*

2. *$\forall K \geq 0, \exists T(\delta, C, K) > 0, \exists D > 0$ such that $\forall \zeta \in Z$, there exists a fundamental solution $\Phi^\zeta$ such that $\forall t \geq T, \forall i \in [3], \|\Phi_{:,i}^\zeta - \hat{\Phi}_{:,i}^{\zeta,K}\| \leq Dt^{Re(R_{ii}^\zeta - K - 1)} e^{Re(\mu_i^\zeta)}$ where $\hat{\Phi}^{\zeta,K} \stackrel{def}{=} \left(I_3 + \sum_{k \leq K} \frac{P_k^\zeta}{t^k}\right) t^{\tilde{R}^\zeta} e^{tA^\zeta}$.*

### H.3 Fundamental solutions around zero and infinity

In this section, we show asymptotic solutions $\tilde{\Phi}$ of the ODE Equation (100) for small and large times. We remind $\gamma_1 = 1$. To get bounded coefficients in the ODE, we will differentiate the two cases $\gamma_2 B\sigma \leq \sqrt{4\gamma_3 B}$ and $\gamma_2 B\sigma \geq \sqrt{4\gamma_3 B}$. The first case corresponds to $\sigma$ small and hence $t$ large with respect to the dimension, where there is acceleration. The second case corresponds to $\sigma$ large, hence $t$ small and dynamics similar to SGD. Additionally, getting asymptotic estimates using Frobenius method around $\infty$ requires the eigenvalues of the leading order matrix to be distinct, since the singularity is of the second kind. Hence we will introduce $\epsilon \in (0, 1)$ and first restrict ourselves to the case where $\gamma_2 B\sigma \notin (1 \pm \epsilon)\sqrt{4\gamma_3 B}$.

**1st case: $\gamma_2 B\sigma \leq \sqrt{4\gamma_3 B}$.** We apply the time change $\tau(t) \stackrel{def}{=} \sigma\sqrt{\gamma_3 B}(1 + t)$ and defining $\hat{\Phi}(\tau) \stackrel{def}{=} \tilde{\Phi}(t)$ we obtain the new ODE

$$\frac{d\hat{\Phi}(\tau)}{d\tau} \stackrel{def}{=} \left(\frac{R}{\sigma\sqrt{\gamma_3 B} + \tau} + A\right)\hat{\Phi}(\tau) = \left(\frac{\begin{pmatrix} 0 & 0 & 0 \\ 0 & -2\delta & 0 \\ 0 & 0 & -\delta \end{pmatrix}}{\sigma\sqrt{\gamma_3 B} + \tau} + \begin{pmatrix} -2\frac{\gamma_2 B\sigma}{\sqrt{\gamma_3 B}} & 0 & -2 \\ 0 & 0 & 2 \\ 1 & -1 & -\frac{\gamma_2 B\sigma}{\sqrt{\gamma_3 B}} \end{pmatrix}\right)\hat{\Phi}(\tau).$$

$$(101)$$

Denoting again $R, A$ respectively the left and right matrices, we rewrite $A = TDT^{-1}$ with $D \stackrel{def}{=} \mathrm{Diag}(\mu_1, \mu_2, \mu_3)$ with $\mu_1 = -\frac{\gamma_2 B\sigma}{\sqrt{\gamma_3 B}} - \sqrt{-4 + \frac{\gamma_2^2 B\sigma^2}{\gamma_3}}$, $\mu_2 = -\frac{\gamma_2 B\sigma}{\sqrt{\gamma_3 B}} + \sqrt{-4 + \frac{\gamma_2^2 B\sigma^2}{\gamma_3}}$, $\mu_3 =$

$-\frac{\gamma_2 B\sigma}{\sqrt{\gamma_3 B}}$. In the limit $\frac{\gamma_2 B\sigma}{\sqrt{\gamma_3 B}} \to 0$, the eigenvalues are distinct. Hence the matrices $T, T^{-1}$ can be chosen analytic in $\frac{\gamma_2 B\sigma}{\sqrt{\gamma_3 B}}$. This implies

$$T = \begin{pmatrix} -i & i & 1 \\ i & -i & 1 \\ 1 & 1 & 0 \end{pmatrix} + \mathcal{O}\left(\frac{\gamma_2 B\sigma}{\sqrt{\gamma_3 B}}\right), \quad T^{-1} = \begin{pmatrix} \frac{i}{4} & -\frac{i}{4} & \frac{1}{2} \\ -\frac{1}{4}i & \frac{i}{4} & \frac{1}{2} \\ \frac{1}{2} & \frac{1}{2} & 0 \end{pmatrix} + \mathcal{O}\left(\frac{\gamma_2 B\sigma}{\sqrt{\gamma_3 B}}\right). \quad (102)$$

Notice that the matrix $A$ has *bounded coefficients* and *distinct eigenvalues bounded away of each other*.
Additionally define $\delta_1 \stackrel{\text{def}}{=} -\delta\left(1 - \frac{\gamma_2 B\sigma}{\sqrt{\gamma_2^2 B^2 \sigma^2 - 4B\gamma_3}}\right)$, $\delta_2 \stackrel{\text{def}}{=} -\delta\left(1 + \frac{\gamma_2 B\sigma}{\sqrt{\gamma_2^2 B^2 \sigma^2 - 4B\gamma_3}}\right)$, $\delta_3 \stackrel{\text{def}}{=} -\delta$.
In the next proposition, we apply Theorem H.2 on a solution $\hat{\Phi}$ of Equation (101) to obtain an asymptotic solution for large time (uniformly in parameter space).

**Proposition H.1** (Asymptotic solutions around infinity). *Let $\epsilon, \tilde{\epsilon} \in (0,1)$, $C > 0$ such that $\delta \in (0, C)$. There exists a constant $M(\epsilon, \tilde{\epsilon}, C) > 0$ such that $\forall B, \gamma_2, \gamma_3, \sigma > 0$ if $\gamma_2 B\sigma < (1-\epsilon)\sqrt{4\gamma_3 B}$, there exists a fundamental solution $\tilde{\Phi}^\infty$ of Equation (100) such that for any $t > -1$ satisfying $\sigma\sqrt{\gamma_3 B}(1+t) > M$*

$$\left\| T^{-1}\tilde{\Phi}^\infty(t) - \left(I_3 + \frac{P_1}{\sigma\sqrt{\gamma_3 B}(1+t)} + \frac{P_2}{\left(\sigma\sqrt{\gamma_3 B}(1+t)\right)^2}\right)\mathcal{D}(\sigma, \gamma_3, B) \right\|$$

$$\leq \frac{\tilde{\epsilon}}{\left(\sigma\sqrt{\gamma_3 B}(1+t)\right)^2}\left(\sigma\sqrt{\gamma_3 B}(1+t)\right)^{-\delta} e^{-\gamma_2 B\sigma^2 t},$$

*where $\mathcal{D}(\sigma, \gamma_3, B) \stackrel{\text{def}}{=} \text{Diag}((\sigma\sqrt{\gamma_3 B}(1+t))^{\delta_i} e^{\sigma\sqrt{\gamma_3 B}\mu_i t}, i = 1, 2, 3)$. Moreover the matrices $P_1, P_2$ are uniquely determined from Theorem H.2.*

Note that above we could directly bound the matrix norm of the difference since the module of the decay is the same on all eigen-vectors ($\text{Re}(\delta_1) = \text{Re}(\delta_2) = \text{Re}(\delta_3)$ and $\text{Re}(\mu_1) = \text{Re}(\mu_2) = \text{Re}(\mu_3)$). Additionally, if $\delta$ is not an integer, we can similarly apply Theorem H.1 to obtain an asymptotic solution for small time (uniformly in parameter space).

**Proposition H.2** (Asymptotic solutions around zero). *Let $\epsilon, \tilde{\epsilon} \in (0,1)$, $C > 0$ such that $\delta \in (0, C)$. There exists a constant $M(\epsilon, \tilde{\epsilon}, C)$ such that $\forall \gamma_2, \gamma_3, \sigma, B > 0$ if $\gamma_2 B\sigma \leq \sqrt{4\gamma_3 B}$ and $d(\delta, \mathbb{N}) > \epsilon$, there exists a fundamental solution $\tilde{\Phi}^0$ of Equation (100) such that for all $t > -1$, if $\sigma\sqrt{\gamma_3 B}(1+t) < M$*

$$\left| \tilde{\Phi}^0(t) - (I_3 + \sigma\sqrt{\gamma_3 B}(1+t)P_1 + \left(\sigma\sqrt{\gamma_3 B}(1+t)\right)^2 P_2) \right.$$

$$\left. \times \begin{pmatrix} 1 & 0 & 0 \\ 0 & ((\sigma\sqrt{\gamma_3 B}(1+t))^{-2\delta} & 0 \\ 0 & 0 & ((\sigma\sqrt{\gamma_3 B}(1+t))^{-\delta} \end{pmatrix} \right|$$

$$\leq \tilde{\epsilon}\left(\sigma\sqrt{\gamma_3 B}(1+t)\right)^2 \mathbb{1}_{3\times 3}\begin{pmatrix} 1 & 0 & 0 \\ 0 & ((\sigma\sqrt{\gamma_3 B}(1+t))^{-2\delta} & 0 \\ 0 & 0 & ((\sigma\sqrt{\gamma_3 B}(1+t))^{-\delta} \end{pmatrix}.$$

$P_1, P_2$ are uniquely determined from Theorem H.1 as

$$P_1 = \begin{pmatrix} -2\frac{\gamma_2 B\sigma}{\sqrt{\gamma_3 B}} & 0 & -\frac{2}{1-\delta} \\ 0 & 0 & \frac{2}{1+\delta} \\ \frac{1}{\delta+1} & -\frac{1}{1-\delta} & -\frac{\gamma_2 B\sigma}{\sqrt{\gamma_3 B}} \end{pmatrix}$$

*and*

$$P_2 = \begin{pmatrix} \frac{1}{2}\left(\frac{4}{x^2} - \frac{2\gamma_1}{\delta+1}\right) & \frac{2}{(2-2\delta)(1-\delta)} & \frac{\frac{4}{(1-\delta)x} + \frac{2}{x}}{2-\delta} \\ \frac{2}{(\delta+1)(2\delta+2)} & -\frac{1}{1-\delta} & -\frac{2}{(\delta+2)x} \\ \frac{-\frac{1}{(\delta+1)x} - \frac{2}{x}}{\delta+2} & \frac{1}{(1-\delta)(2-\delta)x} & \frac{1}{2}\left(-\frac{2}{1-\delta} - \frac{2}{\delta+1} + \frac{1}{x^2}\right) \end{pmatrix}$$

*where $x = \frac{\sqrt{\gamma_3 B}}{\gamma_2 B\sigma}$.*

**2nd Case** $\gamma_2 B\sigma \geq \sqrt{4\gamma_3 B}$. Then we apply the time change $\tau(t) \overset{\text{def}}{=} \gamma_2 B\sigma^2 t$ and define $\hat{\Phi}(\tau) \overset{\text{def}}{=} \tilde{\Phi}(t)$. We obtain the new ODE

$$\frac{\mathrm{d}\hat{\Phi}(\tau)}{\mathrm{d}\tau} = \left( \frac{\begin{pmatrix} 0 & 0 & 0 \\ 0 & -2\delta & 0 \\ 0 & 0 & -\delta \end{pmatrix}}{\gamma_2 B\sigma^2 + \tau} + \begin{pmatrix} -2 & 0 & -2\frac{\sqrt{\gamma_3 B}}{\gamma_2 B\sigma} \\ 0 & 0 & 2\frac{\sqrt{\gamma_3 B}}{\gamma_2 B\sigma} \\ \frac{\sqrt{\gamma_3 B}}{\gamma_2 B\sigma} & -\frac{\sqrt{\gamma_3 B}}{\gamma_2 B\sigma} & -1 \end{pmatrix} \right) \hat{\Phi}(\tau).$$

Again the matrix $A$ has *bounded coefficients* and *distinct eigenvalues bounded away from each other*. We rewrite $A = TDT^{-1}$ with some different matrix $T$ and eigenvalues $\mu_1 = -1 - \sqrt{-4\frac{\gamma_3 B}{\gamma_2^2 B^2 \sigma^2} + 1}$, $\mu_2 = -1 + \sqrt{-4\frac{\gamma_3 B}{\gamma_2^2 B^2 \sigma^2} + 1}$, $\mu_3 = -1$. Denoting $x = \frac{\sqrt{\gamma_3 B}}{\gamma_2 B\sigma}$, we have

$$T = \begin{pmatrix} 1 - x^2 + O\left(x^3\right) & x^2 + O\left(x^3\right) & -2x + O\left(x^3\right) \\ x^2 + O\left(x^3\right) & 1 - x^2 + O\left(x^3\right) & -2x + O\left(x^3\right) \\ -x + O\left(x^3\right) & -x + O\left(x^3\right) & 1 \end{pmatrix},$$

$$T^{-1} = \begin{pmatrix} 1 + 3x^2 + O\left(x^3\right) & x^2 + O\left(x^3\right) & 2x + O\left(x^3\right) \\ x^2 + O\left(x^3\right) & 1 + 3x^2 + O\left(x^3\right) & 2x + O\left(x^3\right) \\ x + O\left(x^3\right) & x + O\left(x^3\right) & 1 + 4x^2 + O\left(x^3\right) \end{pmatrix}.$$

Remind $\delta_1 \overset{\text{def}}{=} -\delta\left(1 - \frac{\gamma_2 B\sigma}{\sqrt{\gamma_2^2 B^2 \sigma^2 - 4\gamma_3 B}}\right)$, $\delta_2 \overset{\text{def}}{=} -\delta\left(1 + \frac{\gamma_2 B\sigma}{\sqrt{\gamma_2^2 B^2 \sigma^2 - 4\gamma_3 B}}\right)$, $\delta_3 \overset{\text{def}}{=} -\delta$. We can again apply Theorem H.1 to obtain that (uniformly in parameter space):

**Proposition H.3** (Asymptotic solutions around infinity). *Let $\epsilon, \tilde{\epsilon} \in (0, 1)$, $C > 0$ such that $\delta, \in (0, C)$. There exists a constant $M(\epsilon, \tilde{\epsilon}, C)$ such that $\forall \gamma_2, \gamma_3, \sigma, t, B > 0$ if $\sigma\sqrt{\gamma_3 B}(1 + t) > M$ and $\gamma_2 B\sigma > (1 + \epsilon)\sqrt{4\gamma_3 B}$, there exists a fundamental solution $\tilde{\Phi}^\infty$ of Equation (100) such that*

$$\left| T^{-1}\tilde{\Phi}^\infty(t) - (I_3 + \frac{P_1}{\gamma_2 B\sigma^2(1 + t)} + \frac{P_2}{(\gamma_2 B\sigma^2(1 + t))^2})\hat{\mathcal{D}} \right| \leq \frac{\tilde{\epsilon}}{(\gamma_2 B\sigma^2(1 + t))^2} \mathbb{1}_{3\times 3} \times \tilde{\mathcal{D}},$$

*where*

$$\hat{\mathcal{D}} \overset{\text{def}}{=} \begin{pmatrix} (\gamma_2 B\sigma^2(1 + t))^{\delta_1} e^{\gamma_2 B\sigma^2 \mu_1 t} & 0 & 0 \\ 0 & (\gamma_2 B\sigma^2(1 + t))^{\delta_2} e^{\gamma_2 B\sigma^2 \mu_2 t} & 0 \\ 0 & 0 & \left(\gamma_2 B\sigma^2(1 + t)\right)^{-\delta_3} e^{\gamma_2 B\sigma^2 \mu_3 t} \end{pmatrix}$$

$$\tilde{\mathcal{D}} \overset{\text{def}}{=} \begin{pmatrix} (\gamma_2 B\sigma^2(1 + t))^{\delta_1} e^{\gamma_2 B\sigma^2 \mu_3 t} & 0 & 0 \\ 0 & (\gamma_2 B\sigma^2(1 + t))^{\delta_2} e^{\gamma_2 B\sigma^2 \mu_2 t} & 0 \\ 0 & 0 & \left(\gamma_2 B\sigma^2(1 + t)\right)^{-\delta_3} e^{\gamma_2 B\sigma^2 \mu_1 t} \end{pmatrix}.$$

*Moreover the matrices $P_1, P_2$ are uniquely determined from Theorem H.2 as*

$$P_1 = \begin{pmatrix} -2 & 0 & \frac{2x}{\delta - 1} \\ 0 & 0 & \frac{2x}{\delta + 1} \\ \frac{x}{\delta + 1} & \frac{x}{\delta - 1} & -1 \end{pmatrix}, \quad P_2 = \begin{pmatrix} \frac{2\delta - x^2 + 2}{\delta + 1} & \frac{x^2}{(\delta - 1)^2} & -\frac{2(\delta - 3)x}{\delta^2 - 3\delta + 2} \\ \frac{x^2}{(\delta + 1)^2} & \frac{x^2}{\delta - 1} & -\frac{2x}{\delta + 2} \\ -\frac{(2\delta + 3)x}{(\delta + 1)(\delta + 2)} & \frac{x}{\delta^2 - 3\delta + 2} & \frac{\delta^2 + 4x^2 - 1}{2(\delta^2 - 1)} \end{pmatrix}.$$

Additionally, if $2\delta$ is not an integer, we can again apply Theorem H.2 to obtain that (uniformly in parameter space):

**Proposition H.4** (Asymptotic solutions around zero). *Let $\epsilon, \tilde{\epsilon} \in (0, 1)$, $C > 0$ such that $\delta \in (0, C)$. There exists a constant $M(\epsilon, \tilde{\epsilon}, C)$ such that $\forall \gamma_2, \gamma_3, \sigma, t, B > 0$ if $\sigma\sqrt{\gamma_3 B}(1 + t) < M$, $\gamma_2 B\sigma \geq \sqrt{4\gamma_3 B}$, and $d(\delta, \mathbb{N}) > \epsilon$, there exists a fundamental solution $\tilde{\Phi}^0$ of Equation (100) such that*

$$\left| \tilde{\Phi}^0(t) - (I_3 + \gamma_2 B\sigma^2(1 + t)P_1 + \left(\gamma_2 B\sigma^2(1 + t)\right)^2 P_2)\hat{\mathcal{D}}, \right| \leq \tilde{\epsilon}\left(\gamma_2 B\sigma^2(1 + t)\right)^2 \mathbb{1}_{3\times 3}\hat{\mathcal{D}}$$

*where*

$$\hat{\mathcal{D}} \overset{def}{=} \begin{pmatrix} 1 & 0 & 0 \\ 0 & ((\gamma_2 B\sigma^2(1+t))^{-2\delta} & 0 \\ 0 & 0 & ((\gamma_2 B\sigma^2(1+t))^{-\delta} \end{pmatrix}.$$

*Moreover the matrices $P_1, P_2$ are uniquely determined from Theorem H.1.*

### H.4  Behavior of $\Phi_{11}(t,s)$ and $\Phi_{12}(t,s)$

The goal of this section is to derive bouds on $\Phi_{11}(t,s)$ and $\Phi_{12}(t,s)$ where $\Phi : \{(t,s) \in (-1,\infty)^2, \ t \geq s\} \to \mathcal{M}_{3\times 3}(\mathbb{R})$ denotes the solution of the IVP (99) with initialization $\Phi(s,s) = I_3$. This implies the initialization condition on Equation (100) that is $\tilde{\Phi}(s,s) = \mathrm{Diag}(1, \frac{\gamma_3}{B\sigma^2}, \frac{\sqrt{\gamma_3}}{\sqrt{B}\sigma})$. We consider several cases.

**1st Case:** $\gamma_2 B\sigma < (1-\epsilon)\sqrt{4\gamma_3 B}$. Depending on the time and $\sigma$, we get different estimates for the values of $\Phi_{11}(t,s)$ and $\Phi_{12}(t,s)$.

For $\sigma\sqrt{\gamma_3 B}(1+t) \leq 1$, we have the following estimate:

**Proposition H.5.** *Let $\epsilon, \tilde{\epsilon} \in (0,1)$, $C > 0$. Let $\delta \in (1,C)$ with $d(\delta, \mathbb{N}) > \epsilon$ and let $\gamma_2, \gamma_3, B, \sigma > 0$, let $0 \leq s \leq t$. Then, there exists a constant $M(\epsilon, \tilde{\epsilon}, C)$ such that if $\sigma\sqrt{\gamma_3 B}(1+t) < M$, and $\gamma_2 B\sigma \leq M\sqrt{4\gamma_3 B}$, the function $\Phi_{11}(t,s) = 1 \pm \tilde{\epsilon}$ and, if additionally $\frac{s}{t} < M$, the function $\Phi_{12}(t,s) \asymp \frac{\gamma_3}{B\sigma^2}(\sigma\sqrt{\gamma_3 B}(1+s))^2 (1 \pm \tilde{\epsilon})$. If $M \leq \sigma\sqrt{\gamma_3 B}(1+t) \leq 1$ or $M\sqrt{4\gamma_3 B} \leq \gamma_2 B\sigma \leq \sqrt{4\gamma_3 B}$ or $M \leq \frac{s}{t} \leq 1$, then $\Phi_{11}(t,s) = \mathcal{O}(1)$ and $\Phi_{12}(t,s) = \mathcal{O}(\frac{\gamma_3}{B\sigma^2}(\sigma\sqrt{\gamma_3 B}(1+s))^2$.*

*Proof.* We apply Proposition H.2 to write that a solution $\Phi$ of Equation (99) is:

$$\Phi(t,s) = \tilde{\Phi}^0(t)\tilde{\Phi}^0(s)^{-1} \mathrm{Diag}(1, \frac{\gamma_3}{B\sigma^2}, \frac{\sqrt{\gamma_3}}{\sqrt{B}\sigma})$$

$$= (I_3 + \sigma\sqrt{\gamma_3 B}(1+t)P_1 + (\sigma\sqrt{\gamma_3 B}(1+t))^2(P_2 \pm \epsilon))$$

$$\times \mathrm{Diag}(1, (\sigma\sqrt{\gamma_3 B}(1+t))^{-2\delta}, (\sigma\sqrt{\gamma_3 B}(1+t))^{-\delta})$$

$$\times \mathrm{Diag}\Big(1, (\sigma\sqrt{\gamma_3 B}(1+s))^{2\delta},$$

$$(\sigma\sqrt{\gamma_3 B}(1+s))^{\delta})(I_3 + \sigma\sqrt{\gamma_3 B}(1+s)P_1 + (\sigma\sqrt{\gamma_3 B}(1+s))^2(P_2 \pm \epsilon)\Big)^{-1}$$

$$\times \mathrm{Diag}\Big(1, \frac{\gamma_3}{B\sigma^2}, \frac{\sqrt{\gamma_3}}{\sqrt{B}\sigma}\Big)$$

$$= \begin{pmatrix} 1 \pm \epsilon & \left(\frac{1+t}{1+s}\right)^{-2\delta}(\sigma\sqrt{\gamma_3 B}(1+t))^2((P_2)_{12} \pm \epsilon) & \left(\frac{1+t}{1+s}\right)^{-\delta}(\sigma\sqrt{\gamma_3 B}(1+t))^1((P_1)_{13} \pm \epsilon) \\ * & * & * \\ * & * & * \end{pmatrix}$$

$$\times \begin{pmatrix} 1 \pm \epsilon & \frac{\gamma_3}{B\sigma^2}(\sigma\sqrt{\gamma_3 B}(1+s))^2\left(((P_1^2 - P_2)_{12} \pm \epsilon\right) & * \\ (\sigma\sqrt{\gamma_3 B}(1+s))^2\left(((P_1^2 - P_2)_{21} \pm \epsilon\right) & \frac{\gamma_3}{B\sigma^2}(1 \pm \epsilon) & * \\ -(\sigma\sqrt{\gamma_3 B}(1+s))^1\left((P_1)_{31} \pm \epsilon\right) & -\frac{\gamma_3}{B\sigma^2}(\sigma\sqrt{\gamma_3 B}(1+s))^1\left(((P_1)_{32} \pm \epsilon\right) & * \end{pmatrix}.$$

Notice that $P_1^2 - P_2 \overset{\frac{\gamma_2 B\sigma}{\sqrt{\gamma_3 B}} \to 0}{\longrightarrow} \begin{pmatrix} \frac{1}{\delta-1} & \frac{1}{(\delta-1)^2} & 0 \\ \frac{1}{(\delta+1)^2} & -\frac{1}{\delta+1} & 0 \\ 0 & 0 & \frac{2}{\delta^2-1} \end{pmatrix}$. Additionally, as $\delta > 1$, we have that

$\left(\frac{1+s}{1+t}\right)^{\delta-1} \ll 1$ for $M$ small. Thus there exists some $M$ small enough such that if $\sigma\sqrt{\gamma_3 B}(1+t) < M$ and $\gamma_2 B\sigma < M\sqrt{4\gamma_3 B}$, then $\Phi_{11}(t,s) = 1 \pm \epsilon$. If, additionally, $\frac{s}{t} \leq M$, then $\Phi_{12}(t,s) \asymp \frac{\gamma_3}{B\sigma^2}(\sigma\sqrt{\gamma_3 B}(1+s))^2 (1 \pm \epsilon)$.

If one of the conditions is fails, we still have that the coefficients of the ODE are bounded for $\sigma\sqrt{\gamma_3 B}(1+t) \leq 1$, $\gamma_2 B\sigma \leq \sqrt{4\gamma_3 B}$, and $\frac{s}{t} \leq 1$. Hence we get under this condition that

$$\boxed{\Phi_{11}(t,s) = \mathcal{O}_+(1) \text{ and } \Phi_{12}(t,s) = \mathcal{O}_+\left(\frac{\gamma_3}{B\sigma^2}(\sigma\sqrt{\gamma_3 B}(1+s))^2\right).}$$ $\qquad\square$

For the case $\sigma\sqrt{\gamma_3 B}(1+t) \geq 1$ and $\sigma\sqrt{\gamma_3 B}(1+s) \leq 1$, we have the following estimate:

**Proposition H.6.** *Let $\epsilon, \tilde{\epsilon} \in (0,1), C > 0$ such that $\delta \in (1,C)$. Let $\gamma_2, \gamma_3, B, \sigma > 0$ and suppose $0 \leq s \leq t$. Moreover suppose that $d(\delta, \mathbb{N}) > \epsilon$ and $\gamma_2 B \sigma < (1-\epsilon)\sqrt{4\gamma_3 B}$. Then there exists a constant $M(\epsilon, \tilde{\epsilon}, C)$ such that if $\sigma\sqrt{\gamma_3 B}(1+t) > M$ and $\sigma\sqrt{\gamma_3 B}(1+t) < \frac{1}{M}$, there exists $C_i, \tilde{C}_i \in \mathbb{R}$ for $i \in [3]$ with $C_1, \tilde{C}_1 > 0$,*

$$\Phi_{11}(t,s) = e^{-\gamma_2 B\sigma^2 t}\left(\sigma\sqrt{\gamma_3 B}(1+t)\right)^{-\delta}$$

$$\times \left[ C_1 + C_2 \cos\left( \log(\sigma\sqrt{\gamma_3 B}(1+t))\frac{\delta\gamma_2 B\sigma}{\sqrt{4\gamma_3 B - \gamma_2^2 B^2 \sigma^2}} + \gamma_2 B\sigma\sqrt{4\gamma_3 B - \gamma_2^2 B^2 \sigma^2 t}\right) \right.$$

$$+ C_3 \sin\left( \log(\sigma\sqrt{\gamma_3 B}(1+t))\frac{\delta\gamma_2 B\sigma}{\sqrt{4\gamma_3 B - \gamma_2^2 B^2 \sigma^2}} + \gamma_2 B\sigma\sqrt{4\gamma_3 B - \gamma_2^2 B^2 \sigma^2 t}\right)$$

$$\left. + \mathcal{O}(\epsilon) + \mathcal{O}(\frac{\gamma_2 B\sigma}{\sqrt{\gamma_3 B}}) \right]$$

*and*

$$\Phi_{12}(t,s) = \frac{\gamma_3}{B\sigma^2}(\sigma\sqrt{\gamma_3 B}(1+s))^2 e^{-\gamma_2 B\sigma^2 t}\left(\sigma\sqrt{\gamma_3 B}(1+t)\right)^{-\delta}$$

$$\times \left[ \tilde{C}_1 + \tilde{C}_2 \cos\left( \log(\sigma\sqrt{\gamma_3 B}(1+t))\frac{\delta\gamma_2 B\sigma}{\sqrt{4\gamma_3 B - \gamma_2^2 B^2 \sigma^2}} + \gamma_2 B\sigma\sqrt{4\gamma_3 B - \gamma_2^2 B^2 \sigma^2 t}\right) \right.$$

$$+ \tilde{C}_3 \sin\left( \log(\sigma\sqrt{\gamma_3 B}(1+t))\frac{\delta\gamma_2 B\sigma}{\sqrt{4\gamma_3 B - \gamma_2^2 B^2 \sigma^2}} + \gamma_2 B\sigma\sqrt{4\gamma_3 B - \gamma_2^2 B^2 \sigma^2 t}\right)$$

$$\left. + \mathcal{O}(\epsilon) + \mathcal{O}(\frac{\gamma_2 B\sigma}{\sqrt{\gamma_3 B}}) \right].$$

*If $\sigma\sqrt{\gamma_3 B}(1+s) \geq \frac{1}{M}$ or $\sigma\sqrt{\gamma_3 B}(1+t) \leq M$ holds, the following is true*

$$\Phi(t,s)_{11} = \mathcal{O}_+\left( (\sigma\sqrt{\gamma_3 B}(1+t))^{-\delta} e^{-\gamma_2 B\sigma^2 t}\right)$$

*and*

$$\Phi(t,s)_{12} = \mathcal{O}_+\left( (\sigma\sqrt{\gamma_3 B}(1+t))^{-\delta} e^{-\gamma_2 B\sigma^2 t}\frac{\gamma_3}{B\sigma^2}(\sigma\sqrt{\gamma_3 B}(1+s))^2\right).$$

*Proof.* We apply Proposition H.2 and Proposition H.1 to decompose

$$\Phi(t,s) = \tilde{\Phi}^\infty(t)\left[\tilde{\Phi}^\infty(1)^{-1}\tilde{\Phi}^0(1)\right]\tilde{\Phi}^0(s)^{-1}\mathrm{Diag}(1, \frac{\gamma_3}{B\sigma^2}, \frac{\sqrt{\gamma_3}}{\sqrt{B}\sigma})$$

$$= \left( \begin{pmatrix} -i & i & 1 \\ i & -i & 1 \\ 1 & 1 & 0 \end{pmatrix} + \mathcal{O}\left(\frac{\gamma_2 B\sigma}{\sqrt{\gamma_3 B}}\right) + \mathcal{O}(\tilde{\epsilon}) \right)(\sigma\sqrt{\gamma_3 B}(1+t))^{-\delta}e^{-\gamma_2 B\sigma^2 t}$$

$$\times \mathrm{Diag}\left( e^{-i\sigma\sqrt{4\gamma_3 B - \gamma_2^2 B^2\sigma^2}t}(\sigma\sqrt{\gamma_3 B}(1+t))^{i\delta\frac{\gamma_2 B\sigma}{\sqrt{4\gamma_3 B - \gamma_2^2 B^2\sigma^2}}}, \right.$$

$$\left. e^{i\sigma\sqrt{4\gamma_3 B - \gamma_2^2 B^2\sigma^2}t}(\sigma\sqrt{\gamma_3 B}(1+t))^{-i\delta\frac{\gamma_2 B\sigma}{\sqrt{4\gamma_3 B - \gamma_2^2 B^2\sigma^2}}}, 1 \right)$$

$$\times \left( S + o_{\frac{\gamma_2 B\sigma}{\sqrt{\gamma_3 B}}+\tilde{\epsilon}}(1) \right) \times \mathrm{Diag}(1, (\sigma\sqrt{\gamma_3 B}(1+s))^{2\delta}, (\sigma\sqrt{\gamma_3 B}(1+s))^\delta)$$

$$\times (I_3 + \sigma\sqrt{\gamma_3 B}(1+s)P_1 + \sigma\sqrt{\gamma_3 B}(1+s)(P_2 \pm \tilde{\epsilon}))^{-1}\mathrm{Diag}\left( 1, \frac{\gamma_3}{B\sigma^2}, \frac{\sqrt{\gamma_3}}{\sqrt{B}\sigma} \right).$$

Here, we used that

$$\hat{\Phi}^\infty(1)^{-1}\hat{\Phi}^0(1) = S + o_{\frac{\gamma_2 B\sigma}{\sqrt{\gamma_3 B}}+\tilde{\epsilon}}(1)$$

where $S$ is a constant matrix. To see that, we need to compare the ODE Equation (101) to the limit ODE with $\frac{\gamma_2 B\sigma}{\sqrt{\gamma_3 B}} \to 0$, $\sigma\sqrt{\gamma_3 B} \to 0$:

$$\frac{\mathrm{d}\tilde{\Phi}(\tau)}{\mathrm{d}\tau} = \left( \frac{\begin{pmatrix} 0 & 0 & 0 \\ 0 & -2\delta & 0 \\ 0 & 0 & -\delta \end{pmatrix}}{\tau} + \begin{pmatrix} 0 & 0 & -2 \\ 0 & 0 & 2 \\ 1 & -1 & 0 \end{pmatrix} \right) \Phi(\tau).$$

Around zero a fundamental solution $j(\tau)$ is asymptotic to $\mathrm{Diag}(1, \tau^{-2\delta}, \tau^{-\delta})$ while around infinity a fundamental solution $\mathsf{j}(\tau)$ is asymptotic to

$$\begin{pmatrix} -i & i & 1 \\ i & -i & 1 \\ 1 & 1 & 0 \end{pmatrix} \mathrm{Diag}(e^{-\gamma_2\sigma\tau}, e^{-\gamma_2\sigma\tau+\sqrt{4\gamma_3}\tau}, e^{-\gamma_2\sigma\tau-\sqrt{4\gamma_3}\tau}).$$

The matrix $S$ is then defined by $S \overset{\text{def}}{=} \mathsf{j}(1)^{-1}j(1)$. We, in particular, know that $\det(S) \neq 0$.

**Lemma H.2.** $\forall t > 0, j(t) \in \mathbb{R}^{3\times 3}$. *Additionally, we have the bounds on the coefficients* $Re(S_{11}) > 0$ *and* $Re(S_{12}) > 0$ *and we can in fact write for some* $C_1^i, C_2^i, C_3^i$ *with* $C_1^i > 0, i \in [2]$:

$$j_1^i(\tau) = \tau^{-\delta}\left( C_1^i + C_2^i \cos\left(\sqrt{4}\tau\right) + C_3^i \sin\left(\sqrt{4}\tau\right) \right) + \mathcal{O}(\tau^{-\delta-1}).$$

*Proof.* Denote $j^i(\tau) = j_{:i}(\tau), \mathsf{j}^i \overset{\text{def}}{=} \mathsf{j}_{:i}(\tau)$ the columns of the fundamental matrices around $0$ and $\infty$, $j, \mathsf{j}$. By definition of $S$, we have for all $\tau > 0$

$$j_1^1(\tau) = \mathsf{j}_1^1(\tau)S_{11} + \mathsf{j}_1^2(\tau)S_{21} + \mathsf{j}_1^3(\tau)S_{31}.$$

By using the asymptotic of $\mathsf{j}$, and the fact that $j_1^1$ must be positive by Corollary C.1 (to better justify because assumption not completely valid, only at the limit), we get that $Re(S_{11}) > 0$. The same argument shows that $Re(S_{12}) > 0$. The last inequality comes by expressing the solution in a $\cos, \sin$ basis. $\qquad\square$

For an estimate of $\Phi_{12}$ we need the 12 coefficient of

$$U \overset{\text{def}}{=} \left( I_3 + \sigma\sqrt{\gamma_3 B}(1+s)P_1 + (\sigma\sqrt{\gamma_3 B}(1+s))^2(P_2 \pm \tilde{\epsilon}) \right)^{-1}.$$

We already did this in the previous proposition. We write

$$U^{-1} = I_3 - \sigma\sqrt{\gamma_3 B}(1+s)P_1 + (\sigma\sqrt{\gamma_3 B}(1+s))^2(P_1^2 - P_2) + \mathcal{O}((\sigma\sqrt{\gamma_3 B}(1+s))^3)$$

and hence

$$\left(U^{-1}\right)_{12} = -(\sigma\sqrt{\gamma_3 B}(1+s))^2 \left( \frac{1}{(\delta-1)^2} \pm \tilde{\epsilon} \right).$$

Finally, if $\delta > 1$ we obtain the result. $\qquad\square$

Next, for $\sigma\sqrt{\gamma_3 B}(1+t) > 1$ and $\sigma\sqrt{\gamma_3 B}(1+s) > 1$, we have the following.

**Proposition H.7.** *Let* $\epsilon, \tilde{\epsilon} \in (0,1), C > 0$ *such that* $\delta \in (1, C)$. *Let* $\gamma_2, \gamma_3, B, \sigma > 0$ *and suppose* $0 \leq s \leq t$. *Moreover suppose that* $\gamma_2 B\sigma < (1-\epsilon)\sqrt{4\gamma_3 B}$. *Then there exists a constant* $M(\epsilon, \tilde{\epsilon}, C)$ *such that if* $\sigma\sqrt{\gamma_3 B}(1+s) > M$,

$$\Phi_{11}(t,s) = \tfrac{1}{2}e^{-\gamma_2 B\sigma^2(t-s)}\left( \tfrac{1+t}{1+s} \right)^{-\delta} \left( 1 + \cos(\sigma\sqrt{4\gamma_3 B}(t-s) + \log\left(\tfrac{1+t}{1+s}\right)\tfrac{\delta\gamma_2 B\sigma}{\sqrt{4\gamma_3 B - \gamma_2^2 B^2\sigma^2}}) + \left( \mathcal{O}\left(\tfrac{\gamma_2 B\sigma}{\sqrt{\gamma_3 B}}\right) + \mathcal{O}(\epsilon) \right) \right)$$

*and*

$$\Phi_{12}(t,s) = \tfrac{\gamma_3}{2B\sigma^2}e^{-\gamma_2 B\sigma^2(t-s)}\left( \tfrac{1+t}{1+s} \right)^{-\delta} \left( 1 - \cos(\sigma\sqrt{4\gamma_3 B}(t-s) + \log\left(\tfrac{1+t}{1+s}\right)\tfrac{\delta\gamma_2 B\sigma}{\sqrt{4\gamma_3 B - \gamma_2^2 B^2\sigma^2}}) + \left( \mathcal{O}\left(\tfrac{\gamma_2 B\sigma}{\sqrt{\gamma_3 B}}\right) + \mathcal{O}(\epsilon) \right) \right).$$

*If, on the other hand,* $\sigma\sqrt{\gamma_3 B}(1+s) \in [1, M]$, *we have*

$$\Phi_{11}(t,s) = \mathcal{O}_+\left( e^{-\gamma_2 B\sigma^2 t}(\sigma\sqrt{\gamma_3 B}(1+t))^{-\delta} \right) \quad \text{and} \quad \Phi_{12}(t,s) = \tfrac{\gamma_3}{B\sigma^2}\mathcal{O}_+\left( e^{-\gamma_2 B\sigma^2 t}(\sigma\sqrt{\gamma_3 B}(1+t))^{-\delta} \right).$$

*Proof.* We use Proposition H.1 to write

$$\Phi(t,s) = \tilde{\Phi}^\infty(t)\tilde{\Phi}^\infty(s)^{-1}\operatorname{Diag}\left(1, \frac{\gamma_3}{B\sigma^2}, \frac{\sqrt{\gamma_3}}{\sqrt{B}\sigma}\right)$$

$$= (T \pm \epsilon)\left(\frac{1+t}{1+s}\right)^{-\delta} e^{-\gamma_2 B\sigma^2 t}$$

$$\times \operatorname{Diag}\left(e^{-i\sigma\sqrt{4\gamma_3 B - \gamma_2^2 B^2\sigma^2}(t-s)}\left(\frac{1+t}{1+s}\right)^{i\delta\frac{\gamma_2 B\sigma}{\sqrt{4\gamma_3 B - \gamma_2^2 B^2\sigma^2}}}, e^{i\sigma\sqrt{4\gamma_3 B - \gamma_2^2 B^2\sigma^2}(t-s)}\left(\frac{1+t}{1+s}\right)^{-i\delta\frac{\gamma_2 B\sigma}{\sqrt{4\gamma_3 B - \gamma_2^2 B^2\sigma^2}}}, 1\right)$$

$$\times (T^{-1} \pm \epsilon) \times \operatorname{Diag}\left(1, \frac{\gamma_3}{B\sigma^2}, \frac{\sqrt{\gamma_3}}{\sqrt{B}\sigma}\right).$$

The same estimate (102) on $T$ we used previously, implies the result. $\qquad\square$

**2nd Case: $\gamma_2 B\sigma > (1+\epsilon)\sqrt{4\gamma_3 B}$.** As in Case 1, we get different estimates for the values of $\Phi_{11}(t,s)$ and $\Phi_{12}(t,s)$.

For $\gamma_2 B\sigma^2(1+t) < 1$, we have the following.

**Proposition H.8.** *Let $\epsilon, \tilde{\epsilon} \in (0,1)$, $C > 0$ such that $\delta \in (1,C)$. Let $\gamma_2, \gamma_3, B, \sigma > 0$, let $0 \le s \le t$. Suppose that $d(\delta, \mathbb{N}) > \epsilon$, $\gamma_2 B\sigma > \sqrt{4\gamma_3 B}$. Then, there exists a constant $M(\epsilon, \tilde{\epsilon}, C)$ such that if $\gamma_2 B\sigma^2(1+t) < M$, $\Phi_{11}(t,s) = 1 \pm \epsilon$ and if additionally $\frac{s}{t} < M$, $\Phi_{12}(t,s) = \mathcal{O}_+\left(\gamma_2^2\right)$. If on the other hand $M \le \gamma_2 B\sigma^2(1+t) \le 1$ or $M \le \frac{s}{t} \le 1$, then $\Phi_{11}(t,s) = \mathcal{O}_+(1)$ and $\Phi_{12}(t,s) = \mathcal{O}_+\left(\gamma_2^2\right)$.*

*Proof.* We use Proposition H.4 to write

$$\Phi(t,s) = \tilde{\Phi}^0(t)\tilde{\Phi}^0(s)^{-1}\operatorname{Diag}\left(1, \frac{\gamma_3}{B\sigma^2}, \frac{\sqrt{\gamma_3}}{\sqrt{B}\sigma}\right)$$

$$= (I_3 + (\gamma_2 B\sigma^2(1+t))P_1 + (\gamma_2 B\sigma^2(1+t))^2(P_2 \pm \epsilon))\begin{pmatrix} 1 & 0 & 0 \\ 0 & \left(\frac{1+t}{1+s}\right)^{-2\delta} & 0 \\ 0 & 0 & \left(\frac{1+t}{1+s}\right)^{-\delta} \end{pmatrix}$$

$$\times (I_3 + (\gamma_2 B\sigma^2(1+s))P_1 + (\gamma_2 B\sigma^2(1+s))^2(P_2 \pm \epsilon))^{-1}\operatorname{Diag}\left(1, \frac{\gamma_3}{B\sigma^2}, \frac{\sqrt{\gamma_3}}{\sqrt{B}\sigma}\right)$$

$$= \begin{pmatrix} 1 \pm \epsilon & (\gamma_2 B\sigma^2(1+t))^2((P_2)_{12} \pm \epsilon) & (\gamma_2 B\sigma^2(1+t))((P_1)_{13} \pm \epsilon) \\ * & * & * \\ * & * & * \end{pmatrix}\begin{pmatrix} 1 & 0 & 0 \\ 0 & \left(\frac{1+t}{1+s}\right)^{-2\delta} & 0 \\ 0 & 0 & \left(\frac{1+t}{1+s}\right)^{-\delta} \end{pmatrix}$$

$$\times \begin{pmatrix} 1 \pm \epsilon & (\gamma_2 B\sigma^2(1+s))^2((P_1^2 - P_2)_{12} \pm \epsilon) & * \\ (\gamma_2 B\sigma^2(1+s))^2((P_1^2 - P_2)_{21} \pm \epsilon) & 1 \pm \epsilon & * \\ (\gamma_2 B\sigma^2(1+s))(-(P_1)_{31} \pm \tilde{\epsilon}) & (\gamma_2 B\sigma^2(1+s))(-(P_1)_{32} \pm \tilde{\epsilon}) & * \end{pmatrix}\operatorname{Diag}\left(1, \frac{\gamma_3}{B\sigma^2}, \frac{\sqrt{\gamma_3}}{\sqrt{B}\sigma}\right).$$

Note that $(P_2)_{12}, (P_1)_{12}^2 = \mathcal{O}\left(\frac{\sqrt{\gamma_3 B}}{\gamma_2 B\sigma}\right)^2$ and $(P_1)_{13}, (P_1)_{32} = \mathcal{O}\left(\frac{\sqrt{\gamma_3 B}}{\gamma_2 B\sigma}\right)$. Hence we obtain

$$\boxed{\Phi_{11}(t,s) = 1 \pm \epsilon \text{ and } \Phi_{12}(t,s) = \mathcal{O}_+\left(\frac{\gamma_3}{B\sigma^2}(\gamma_2 B\sigma^2(1+s))^2\left(\left(\frac{\sqrt{\gamma_3 B}}{\gamma_2 B\sigma}\right)^2 \pm \epsilon\right)\right) = \mathcal{O}_+(\gamma_2^2).}$$

Additionally if $\gamma_2 B\sigma^2(1+t) \in [M,1]$ we still have the bounds: $\Phi_{11}(t,s) = \mathcal{O}_+(1)$ and $\Phi_{12}(t,s) = \mathcal{O}_+\left(\gamma_2^2\right)$. $\qquad\square$

For $\gamma_2 B\sigma^2(1+t) > 1$ and $\gamma_2 B\sigma^2(1+s) < 1$, we have the following.

**Proposition H.9.** *Let $\epsilon \in (0,1)$, $C > 0$ such that $\delta \in (1,C)$. Let $\gamma_2, \gamma_3, B, \sigma > 0$ and suppose $0 \le s \le t$. Moreover, suppose that $\gamma_2 B\sigma > (1+\epsilon)\sqrt{4\gamma_3 B}$, $d(\delta, \mathbb{N}) > \epsilon)$, $\gamma_2 B\sigma^2(1+s) \le 1$, and $\gamma_2 B\sigma^2(1+t) \ge 1$, then*

$$\boxed{\Phi_{11}(t,s) = \mathcal{O}_+\left(e^{-\gamma_2 B\sigma^2(1+t)} + \left(\gamma_2 B\sigma^2(1+t)\right)^{-\delta}\right)}$$

*and*

$$\Phi_{12}(t,s) = \mathcal{O}_+ \left( \frac{\gamma_3}{B\sigma^2} \left( \gamma_2 B\sigma^2(1+s) \right)^2 \left( e^{-\gamma_2 B\sigma^2(1+t)} + \left( \gamma_2 B\sigma^2(1+t) \right)^{-\delta} \right) \right).$$

*Proof.* We use Proposition H.3 and Proposition H.4 to decompose

$$\Phi(t,s) = \tilde{\Phi}^\infty(t) \left[ \tilde{\Phi}^\infty(1)^{-1} \tilde{\Phi}^0(1) \right]^{-1} \tilde{\Phi}^0(s)^{-1} \operatorname{Diag}(1, \frac{\gamma_3}{B\sigma^2}, \frac{\sqrt{\gamma_3}}{\sqrt{B}\sigma})$$

$$= T(I_3 + (\gamma_2 B\sigma^2(1+t))^{-1} P_1 + (\gamma_2 B\sigma^2(1+t))^{-2}(P_2 \pm \epsilon))$$

$$\cdot \operatorname{Diag}(e^{\mu_1 t}(\gamma_2 B\sigma^2(1+t))^{\delta_1}, e^{\mu_2 t}(\gamma_2 B\sigma^2(1+t))^{\delta_2}, e^{\mu_3 t}(\gamma_2 B\sigma^2(1+t))^{\delta_3})$$

$$\cdot \left( D + o_{\frac{\sqrt{\gamma_3 B}}{\gamma_2 B\sigma}}(1) \right)$$

$$\cdot \operatorname{Diag}(1, (\gamma_2 B\sigma^2(1+s))^{2\delta}, (\gamma_2 B\sigma^2(1+s))^{\delta})$$

$$\cdot (I_3 + (\gamma_2 B\sigma^2(1+s)) P_1 + (\gamma_2 B\sigma^2(1+s))^2 (P_2 \pm \epsilon))^{-1}$$

$$\cdot \operatorname{Diag}(1, \frac{\gamma_3}{B\sigma^2}, \frac{\sqrt{\gamma_3}}{\sqrt{B}\sigma}).$$

Here we have written $\left[ \hat{\Phi}^\infty(1)^{-1} \hat{\Phi}^0(1) \right]^{-1} = D + o_{\frac{\sqrt{\gamma_3 B}}{\gamma_2 B\sigma}}(1))$ where $D = \operatorname{Diag}(d_1, d_2, d_3)$ for $d_i > 0, i \in [3]$. To see that compare with the limit of ODE Equation (100) as $\frac{\sqrt{\gamma_3 B}}{\gamma_2 B\sigma} \to 0, \gamma_2 B\sigma^2 \to 0$:

$$\frac{d\hat{\Phi}(\tau)}{d\tau} = \left( \frac{\begin{pmatrix} 0 & 0 & 0 \\ 0 & -2\delta & 0 \\ 0 & 0 & -\delta \end{pmatrix}}{\tau} + \begin{pmatrix} -2 & 0 & 0 \\ 0 & 0 & 0 \\ 0 & 0 & -1 \end{pmatrix} \right) \hat{\Phi}(\tau).$$

We know that around $\infty$ there is a fundamental solution $\mathsf{j}$ asymptotic to $\operatorname{Diag}(e^{-2\tau}, 1, e^{-\tau})$ and around $0$ there is a fundamental solution $j$ asymptotic to $\operatorname{Diag}(1, \tau^{-2\delta}, \tau^{-\delta})$. It is clear that in fact both solutions are diagonal, positive and hence we can define the positive diagonal matrix $D = \mathsf{j}(1)^{-1} j(1)$. This implies directly the bound on $\Phi_{11}$ as

$$\Phi_{11}(t,s) = \gamma_2^2 B\mathcal{O} \left( e^{\mu_1 t}(1+t)^{\delta_1} + e^{\mu_2 t}(1+t)^{\delta_2} + e^{\mu_3 t}(1+t)^{\delta_3} \right)$$

$$= \gamma_2^2 \mathcal{O}_+ (e^{-\gamma_2 B\sigma^2 t} + (\gamma_2 B\sigma^2 t)^{-\delta})$$

where we used that $\mu_1 \le \mu_2 = -\gamma_2 B\sigma^2 \le \mu_3 \le 0$ and $\delta_2 \le \delta_3 = -\delta \le 0$ with $\delta_1$ bounded above by some constant of $\epsilon, \delta$. For the bound on $\Phi_{12}$, notice additionally that since $\delta > 1$, the term $\left( \gamma_2 B\sigma^2(1+s) \right)^2 \ge \left( \gamma_2 B\sigma^2(1+s) \right)^{\delta+1} \ge \left( \gamma_2 B\sigma^2(1+s) \right)^{2\delta}$. $\square$

For $\gamma_2 B\sigma^2(1+t) > 1$ and $\gamma_2 B\sigma^2(1+s) > 1$, we have the following

**Proposition H.10.** *Let $\epsilon \in (0,1)$, $C > 0$ such that $\delta \in (1, C)$. Let $\gamma_2, \gamma_3, B, \sigma > 0$, let $0 \le s \le t$. Suppose that $\gamma_2 B\sigma^2(1+s) > 1$, $\gamma_2 B\sigma > (1+\epsilon)\sqrt{4\gamma_3 B}$, then*

$$\Phi_{11}(t,s) = (1 + \mathcal{O}((\gamma_2 B\sigma^2(1+s))^{-1} + x)e^{\mu_1(t-s)} \left( \frac{1+t}{1+s} \right)^{\delta_1}$$

$$+ \mathcal{O}(x^4 + x^2(\gamma_2 B\sigma^2(1+s))^{-2} + (\gamma_2 B\sigma^2(1+t))^{-2})e^{\mu_2(t-s)} \left( \frac{1+t}{1+s} \right)^{\delta_2}$$

$$+ \mathcal{O}(x^2 + x(\gamma_2 B\sigma^2(1+s))^{-1} + (\gamma_2 B\sigma^2(1+t))^{-2})e^{\mu_3(t-s)} \left( \frac{1+t}{1+s} \right)^{\delta_3},$$

*and*

$$\Phi_{12}(t,s) = \frac{\gamma_3}{B\sigma^2}\Bigg(\mathcal{O}(x^2 + (\gamma_2 B\sigma^2(1+s))^{-2})e^{\mu_1(t-s)}\left(\frac{1+t}{1+s}\right)^{\delta_1}$$

$$+ \mathcal{O}(x^2 + (\gamma_2 B\sigma^2(1+t))^{-2})e^{\mu_2(t-s)}\left(\frac{1+t}{1+s}\right)^{\delta_2}$$

$$+ \mathcal{O}(x^2 + x(\gamma_2 B\sigma^2(1+s))^{-1} + (\gamma_2 B\sigma^2(1+t))^{-2})e^{\mu_3(t-s)}\left(\frac{1+t}{1+s}\right)^{\delta_3}\Bigg),$$

*where* $x = \frac{\sqrt{\gamma_3 B}}{\gamma_2 B\sigma}$.

*Proof.* We use Proposition H.3 to decompose

$$\Phi(t,s) = \tilde{\Phi}^\infty(t)\tilde{\Phi}^\infty(s)^{-1}\,\text{Diag}\left(1, \frac{\gamma_3}{B\sigma^2}, \frac{\sqrt{\gamma_3}}{\sqrt{B}\sigma}\right)$$

$$= T(I_3 + (\gamma_2 B\sigma^2(1+t))^{-1}P_1 + (\gamma_2 B\sigma^2(1+t))^{-2}(P_2 \pm \epsilon))$$

$$\cdot \begin{pmatrix} e^{\mu_1(t-s)}\left(\frac{1+t}{1+s}\right)^{\delta_1} & 0 & 0 \\ 0 & e^{\mu_2(t-s)}\left(\frac{1+t}{1+s}\right)^{\delta_2} & 0 \\ 0 & 0 & e^{\mu_3(t-s)}\left(\frac{1+t}{1+s}\right)^{\delta_3} \end{pmatrix}$$

$$\cdot (I_3 + (\gamma_2 B\sigma^2(1+s))^{-1}P_1 + (\gamma_2 B\sigma^2(1+s))^{-2}(P_2 \pm \epsilon))^{-1}T^{-1}\,\text{Diag}(1, \frac{\gamma_3}{B\sigma^2}, \frac{\sqrt{\gamma_3}}{\sqrt{B}\sigma})$$

$$= \begin{pmatrix} 1 + \mathcal{O}(x + \frac{1}{\gamma_2 B\sigma^2 t}) & \mathcal{O}(x^2 + \left(\frac{1}{\gamma_2 B\sigma^2 t}\right)^2) & \mathcal{O}(x + \left(\frac{1}{\gamma_2 B\sigma^2 t}\right)^2) \\ * & * & * \\ * & * & * \end{pmatrix}$$

$$\cdot \begin{pmatrix} e^{\mu_1(t-s)}\left(\frac{1+t}{1+s}\right)^{\delta_1} & 0 & 0 \\ 0 & e^{\mu_2(t-s)}\left(\frac{1+t}{1+s}\right)^{\delta_2} & 0 \\ 0 & 0 & e^{\mu_3(t-s)}\left(\frac{1+t}{1+s}\right)^{\delta_3} \end{pmatrix}$$

$$\cdot \begin{pmatrix} 1 + \mathcal{O}(x + \frac{1}{\gamma_2 B\sigma^2 s}) & \mathcal{O}(x^2 + \left(\frac{1}{\gamma_2 B\sigma^2 s}\right)^2) & * \\ \mathcal{O}(x^2 + \left(\frac{1}{\gamma_2 B\sigma^2 s}\right)^2) & 1 + \mathcal{O}(x + \frac{1}{\gamma_2 B\sigma^2 s}) & * \\ \mathcal{O}(x + \left(\frac{1}{\gamma_2 B\sigma^2 s}\right))^2 & \mathcal{O}(x + \left(\frac{1}{\gamma_2 B\sigma^2 s}\right)^2) & * \end{pmatrix}\,\text{Diag}(1, \frac{\gamma_3}{B\sigma^2}, \frac{\sqrt{\gamma_3}}{\sqrt{B}\sigma})$$

where we used the estimate (102) for $T$ and (H.4) for $P_1, P_2$. We additionally remind $\mu_1 = -\gamma_2 B\sigma^2 - \sigma\sqrt{\gamma_2^2 B^2\sigma^2 - 4\gamma_3 B}$, $\mu_2 = -\gamma_2 B\sigma^2 + \sigma\sqrt{\gamma_2^2 B^2\sigma^2 - 4\gamma_3 B} \asymp -2\frac{\gamma_3}{\gamma_2}$, $\mu_3 = -\gamma_2 B\sigma^2$ and $\delta_1 = -\delta\left(1 - \frac{\gamma_2 B\sigma}{\sqrt{\gamma_2^2 B^2\sigma^2 - 4\gamma_3 B}}\right)$, $\delta_2 = -\delta\left(1 + \frac{\gamma_2 B\sigma}{\sqrt{\gamma_2^2 B^2\sigma^2 - 4\gamma_3 B}}\right)$, $\delta_3 = -\delta$. Especially we have for $\gamma_2 B\sigma > (1+\epsilon)\sqrt{4\gamma_3 B}$ that $\mu_1 \asymp -2\gamma_2 B\sigma^2$, $\mu_2 \asymp -\frac{2\gamma_3}{\gamma_2}$, and $\delta_1 \asymp -\delta\frac{2\gamma_3 B}{\gamma_2^2 B^2\sigma^2}$, $\delta_2 \asymp -2\delta$. Since $\mu_1, \mu_2, \mu_3 \lesssim -\frac{1}{d}$, this directly implies that $\Phi_{11}, \Phi_{12}(t,s) = \mathcal{O}_+\left(e^{-\frac{t-s}{d}}\right)$. Ultimately, this leads to

$$\Phi_{11}(t,s) = (1 + \mathcal{O}((\gamma_2 B\sigma^2(1+s))^{-1} + x)e^{\mu_1(t-s)}\left(\frac{1+t}{1+s}\right)^{\delta_1}$$

$$+ \mathcal{O}(x^4 + x^2(\gamma_2 B\sigma^2(1+s))^{-2} + (\gamma_2 B\sigma^2(1+t))^{-2})e^{\mu_2(t-s)}\left(\frac{1+t}{1+s}\right)^{\delta_2} \quad (103)$$

$$+ \mathcal{O}(x^2 + x(\gamma_2 B\sigma^2(1+s))^{-1} + (\gamma_2 B\sigma^2(1+t))^{-2})e^{\mu_3(t-s)}\left(\frac{1+t}{1+s}\right)^{\delta_3},$$

and

$$\Phi_{12}(t,s) = \frac{\gamma_3}{B\sigma^2}\left(\mathcal{O}(x^2 + (\gamma_2 B\sigma^2(1+s))^{-2})e^{\mu_1(t-s)}\left(\frac{1+t}{1+s}\right)^{\delta_1}\right.$$

$$+ \mathcal{O}(x^2 + (\gamma_2 B\sigma^2(1+t))^{-2})e^{\mu_2(t-s)}\left(\frac{1+t}{1+s}\right)^{\delta_2}$$

$$+ \mathcal{O}(x^2 + x(\gamma_2 B\sigma^2(1+s))^{-1}$$

$$\left.+ (\gamma_2 B\sigma^2(1+t))^{-2})e^{\mu_3(t-s)}\left(\frac{1+t}{1+s}\right)^{\delta_3}\right).$$

(104)

$\square$

## H.5 Summary

We informally summarize all the bounds on $\Phi_{11}(t,s)$, $\Phi_{12}(t,s)$ that we previously developed. In red, we outline the important contributions for the first and second terms of the kernel.

- $\Phi_{\sigma^2}(t,s)_{11}$, $s \le t \le \frac{\gamma_2}{\gamma_3}$ .

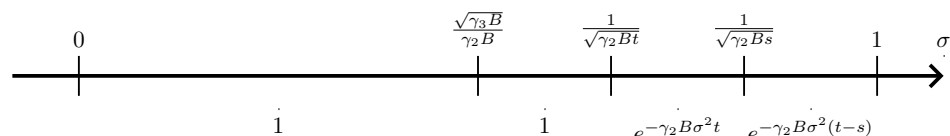

- $\Phi_{\sigma^2}(t,s)_{11}$, $s \le \frac{\gamma_2}{\gamma_3} \le t$ .

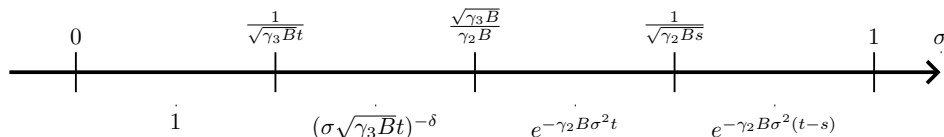

- $\Phi_{\sigma^2}(t,s)_{11}$, $\frac{\gamma_2}{\gamma_3} \le s \le t$ .

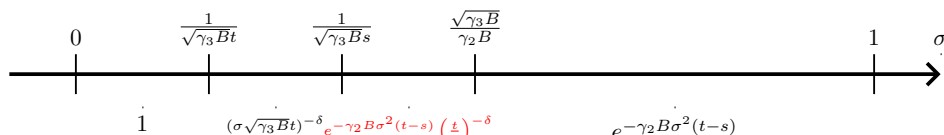

- $\Phi_{\sigma^2}(t,s)_{12}$, $s \le t \le \frac{\gamma_2}{\gamma_3}$ .

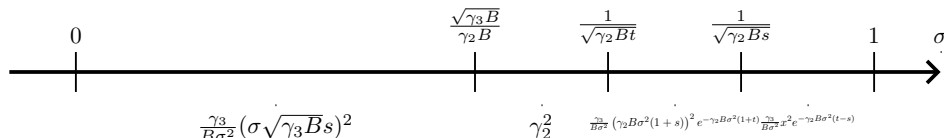

- $\Phi_{\sigma^2}(t,s)_{12}$, $s \le \frac{\gamma_2}{\gamma_3} \le t$ .

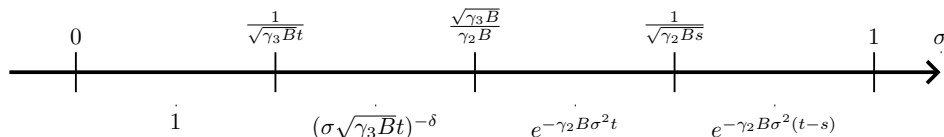

- $\Phi_{\sigma^2}(t,s)_{12}$, $\frac{\gamma_2}{\gamma_3} \le s \le t$ .

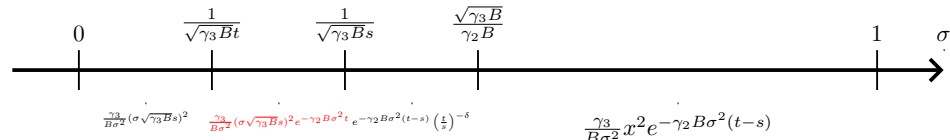

## H.6 Bounds at the singular point $\gamma_2 B\sigma \in [(1-\epsilon)\sqrt{4\gamma_3 B}, (1+\epsilon)\sqrt{4\gamma_3 B}]$

Denote $\omega \stackrel{\text{def}}{=} \frac{\gamma_2 B\sigma}{\sqrt{4\gamma_3 B}} - 1$. For $\omega$ bounded away from zero, we derived in propositions H.1 and H.3 asymptotics solutions for large time of Equation (99). We still need to do the same for $\omega \approx 0$. To that end, we will show that the solutions constructed near zero in Propositions H.2 and H.4 have exponential decay for large time. We will follow similar steps as in [78, Lemma D.3].

The goal is to bound $\Phi_{11}(t, s), \Phi_{12}(t, s)$ where $\Phi(t, s)$ satisfies the IVP Equation (99) with $\Phi(s, s) = I_3$. Denote

$$S = \begin{pmatrix} -1 & \frac{1}{2} & -\frac{1}{4} \\ -1 & -\frac{1}{2} & -\frac{1}{4} \\ 1 & 0 & 0 \end{pmatrix},\tag{105}$$

then we have the identities

$$S^{-1}\left(\begin{pmatrix} -2\gamma_2 B\sigma & 0 & -2\sqrt{\gamma_3 B} \\ 0 & 0 & 2\sqrt{\gamma_3 B} \\ \sqrt{\gamma_3 B} & -\sqrt{\gamma_3 B} & -\gamma_2 B\sigma \end{pmatrix} + \gamma_2 B\sigma I_3\right)S = \begin{pmatrix} 0 & \sqrt{\gamma_3 B} & 0 \\ 0 & 0 & \sqrt{\gamma_3 B} \\ 0 & 0 & 0 \end{pmatrix}$$

$$+ \omega\begin{pmatrix} 0 & 0 & 0 \\ 4\sqrt{\gamma_3 B} & 0 & \sqrt{\gamma_3 B} \\ 0 & 4\sqrt{\gamma_3 B} & 0 \end{pmatrix},$$

and

$$S^{-1}\begin{pmatrix} 0 & 0 & 0 \\ 0 & -2\delta & 0 \\ 0 & 0 & -\delta \end{pmatrix}S = \begin{pmatrix} -\delta & 0 & 0 \\ -2\delta & -\delta & -\frac{\delta}{2} \\ 0 & -2\delta & -\delta \end{pmatrix}.$$

Then (99) can be rewritten as

$$\frac{\mathrm{d}\hat{\Phi}(t)}{\mathrm{d}t} = \left(\frac{\begin{pmatrix} -\delta & 0 & 0 \\ -2\delta & -\delta & -\frac{\delta}{2} \\ 0 & -2\delta & -\delta \end{pmatrix}}{\sigma\sqrt{\gamma_3 B} + t} + \begin{pmatrix} 0 & 1 & 0 \\ 0 & 0 & 1 \\ 0 & 0 & 0 \end{pmatrix} + \omega\begin{pmatrix} 0 & 0 & 0 \\ 4 & 0 & 1 \\ 0 & 4 & 0 \end{pmatrix}\right)\hat{\Phi}(t)\tag{106}$$

where: $\hat{\Phi}(t) \stackrel{\text{def}}{=} S\begin{pmatrix} \Phi_1\left(\frac{t}{\sigma\sqrt{\gamma_3 B}}\right) \\ \frac{\gamma_3}{B\sigma^2}\Phi_2\left(\frac{t}{\sigma\sqrt{\gamma_3 B}}\right) \\ \frac{\sqrt{\gamma_3}}{\sqrt{B}\sigma}\Phi_3\left(\frac{t}{\sigma\sqrt{\gamma_3 B}}\right) \end{pmatrix} e^{\gamma_2 B\sigma^2 t}$. Now define $\xi^2(t) \stackrel{\text{def}}{=} \max\{\omega, \frac{1}{\sigma\sqrt{\gamma_3 B}+t}\}$

We then have:

$$\begin{pmatrix} \hat{\Phi}'_1(t) \\ \xi^{-1}\hat{\Phi}'_2(t) \\ \xi^{-2}\hat{\Phi}'_3(t) \end{pmatrix} = \left(\frac{\begin{pmatrix} -\delta & 0 & 0 \\ -2\xi^{-1}\delta & -\delta & -\frac{\delta}{2}\xi \\ 0 & -2\delta\xi^{-1} & -\delta \end{pmatrix}}{\sigma\sqrt{\gamma_3 B} + t} + \begin{pmatrix} 0 & \xi & 0 \\ 0 & 0 & \xi \\ 0 & 0 & 0 \end{pmatrix}\right.$$

$$\left. + \omega\begin{pmatrix} 0 & 0 & 0 \\ 4\xi^{-1} & 0 & \xi \\ 0 & 4\xi^{-1} & 0 \end{pmatrix}\right)\begin{pmatrix} \hat{\Phi}_1(t) \\ \xi^{-1}\hat{\Phi}_2(t) \\ \xi^{-2}\hat{\Phi}_3(t) \end{pmatrix}.$$

Denote $N(t) = \left\| \begin{pmatrix} \hat{\Phi}_1(t) \\ \xi^{-1}\hat{\Phi}_2(t) \\ \xi^{-2}\hat{\Phi}_3(t) \end{pmatrix} \right\|_\infty$ and $A(t)$ the operator norm for the infinity norm of the right-hand matrix

$$A(t) \stackrel{\text{def}}{=} \left\| \frac{\begin{pmatrix} -\delta & 0 & 0 \\ -2\xi^{-1}\delta & -\delta & -\frac{\delta}{2}\xi \\ 0 & -2\delta\xi^{-1} & -\delta \end{pmatrix}}{\sigma\sqrt{\gamma_3 B} + t} + \begin{pmatrix} 0 & \xi & 0 \\ 0 & 0 & \xi \\ 0 & 0 & 0 \end{pmatrix} + \omega \begin{pmatrix} 0 & 0 & 0 \\ 4\xi^{-1} & 0 & \xi \\ 0 & 4\xi^{-1} & 0 \end{pmatrix} \right\|_\infty .$$

Then we have the identity:

$$N'(t) \le A(t)N(t) + \mathbb{1}_{\xi^2 = \frac{1}{\sigma\sqrt{\gamma_3 B} + t}} \frac{N(t)}{\sigma\sqrt{\gamma_3 B} + t}.$$

From the above, if $|\omega| < 1$, we know the existence of a continuous constant $M(\delta) > 0$ such that $\forall t \ge 0$ with $\sigma\sqrt{\gamma_3 B} + t \ge 1$, $A(t) \le M\xi(t)$.

By Gronwall's lemma, it implies that for all $t \ge s$ with $\sigma\sqrt{\gamma_3 B} + s \ge 1$,

$$N(t) \le N(s)e^{(M+1)\int_s^t \xi(s)\, \mathrm{d}s}. \tag{107}$$

**Proposition H.11.** *Denote* $\omega \stackrel{\text{def}}{=} \frac{\gamma_2 B\sigma}{\sqrt{4\gamma_3 B}} - 1$. *There exists some* $c < 1$ *and some continuous function* $M(\delta)$ *such that if* $|\omega| < c$ *and* $t \ge s \ge 0$,

$$\Phi(t,s)_{11} \le Me^{-\frac{\gamma_2 B\sigma^2}{2}(t-s)}, \quad \Phi(t,s)_{12} \le \frac{\gamma_3}{B\sigma^2}Me^{-\frac{\gamma_2 B\sigma^2}{2}(t-s)}.$$

*Proof.* The proof is similar to the one of [78, Lemma D.3].

If $\sigma\sqrt{\gamma_3 B}(1+t) \ge \sigma\sqrt{\gamma_3 B}(1+s) \ge 1$ we can apply Equation (107) to each column of $\Phi(t,s)$. For the first column, we know that $N(s) \lesssim \xi(s)^{-2}$ and $\int_s^t \xi(s)\,\mathrm{d}s \lesssim \xi(t)(t-s)$ since $\xi(t)$ follows a power law. This brings that $\forall t \ge s$, $|\hat{\Phi}_3(t)| \lesssim \xi(t)^2/\xi(s)^2 e^{(t-s)\xi(t)} \le e^{(t-s)\xi(t)}$. We now repeat the same procedure on the vector $\begin{pmatrix} \hat{\Phi}_1(t) \\ \xi^{-1}\hat{\Phi}_2(t) \end{pmatrix}$ to obtain similarly $|\hat{\Phi}_2(t)| \lesssim \xi(t)/\xi(s)e^{(t-s)\xi(t)} \le e^{(t-s)\xi(t)}$ and finally on the single vector $(\hat{\Phi}_1(t))$ to obtain $|\hat{\Phi}_2(t)| \lesssim e^{(t-s)\xi(t)}$. Notice that $(t-s)\xi(t) \lesssim \max\{\sqrt{|\omega|}, \sqrt{\sigma\sqrt{\gamma_3 B}t}\}$. This all bring $\Phi(t,s)_{11} \le Me^{-\gamma_2 B\sigma^2(1-M\sqrt{|\omega|})(t-s)}$, $\Phi(t,s)_{12} \le \frac{\gamma_3}{B\sigma^2}Me^{-\gamma_2 B\sigma^2(1-M\sqrt{|\omega|})(t-s)}$. For the second column notice the additional factor $\frac{\gamma_3}{B\sigma^2}$. Finally, to extend to $\sigma\sqrt{\gamma_3 B}(1+s) \le 1$, notice that we still have $\|\Phi(1,s)\| \lesssim 1$ and we can apply the same argument for $\sigma\sqrt{\gamma_3 B}(1+t) \ge 1$. For $\sigma\sqrt{\gamma_3 B}(1+t) \le 1$ this is also clear since $\|\Phi(t,s)\| \lesssim 1$ $\qquad \square$

### H.7 Estimation of the forcing function

In the following section, we will estimate the three forcing terms $\mathcal{F}_0, \mathcal{F}_{pp}, \mathcal{F}_{ac}$.

#### H.7.1 Asymptotics of $\mathcal{F}_0(t)$

Remind the definition

$$\mathcal{F}_0(t) \stackrel{\text{def}}{=} \mu_{\mathcal{F}_0}(\{0\})(\Phi^{\sigma=0}(t,0))_{11}.$$

Then we have the result from [80]:

| $\sigma \leq \frac{\sqrt{\gamma_3 B}}{\gamma_2 B}$ | | $\sigma \geq \frac{\sqrt{\gamma_3 B}}{\gamma_2 B}$ | |
|---|---|---|---|
| $\sigma < \frac{1}{\sqrt{\gamma_3 B t}}$ | $\frac{1}{\sqrt{\gamma_3 B t}} < \sigma$ 
 $(\implies t > \frac{\gamma_2}{\gamma_3})$ | $\sigma < \frac{1}{\sqrt{\gamma_2 B t}}$ 
 $(\implies t < \frac{\gamma_2}{\gamma_3})$ | $\frac{1}{\sqrt{\gamma_2 B t}} < \sigma$ |

Table 9: Cuts of $\sigma$ domains for forcing function. This is a simplification of Table 11 using $s = 0$.

**Proposition H.12.** *Suppose $\frac{v}{d} > 1$, $\alpha > 0$ and $2\alpha + 2\beta > 1$. then the constant function $\mathcal{F}_0(t)$ satisfies:*

$$\mathcal{F}_0(t) \asymp d^{-2\alpha + \max\{0, 1-2\beta\}}.$$

*Proof.* From [80] or proposition D.3, we can rewrite

$$\mathcal{F}_0(t) = \sum_{j=1}^{v} \frac{j^{-2\alpha-2\beta}}{1 + j^{-2\alpha} d^{2\alpha} \kappa(v/d)} \left(1 + \mathcal{O}(d^{-1})\right) \text{ where } \kappa(v/d) \text{ solves } \int_0^{v/d} \frac{\kappa}{\kappa + u^{2\alpha}} \, \mathrm{d}u = 1.$$

For large $d$, the result was proven in [80, Lemma H.3]. For small $d$ just notice that $\mathcal{F}_0(t) > 0$. □

### H.7.2 Asymptotics of $\mathcal{F}_{ac}$ and $\mathcal{F}_{pp}$

We remind the definitions of the pure-point term and absolutely continuous term of the forcing function

$$\mathcal{F}_{pp}(t) \overset{\text{def}}{=} \int_0^1 (\sigma^2)^{\frac{2\beta-1}{2\alpha}} \Phi_{11}^\sigma(t, 0) \, \mathrm{d}(\sigma^2) = 2 \int_0^1 \sigma^{1 + \frac{2\beta-1}{\alpha}} \Phi_{11}^\sigma(t, 0) \, \mathrm{d}\sigma,$$

$$\mathcal{F}_{ac}(t) \overset{\text{def}}{=} \int_0^1 (\sigma^2)^{-\frac{1}{2\alpha}} \Phi_{11}^\sigma(t, 0) \, \mathrm{d}(\sigma^2) = 2c_\beta \int_{d^{-\alpha}}^1 \sigma^{1 - \frac{1}{\alpha}} \Phi_{11}^\sigma(t, 0) \, \mathrm{d}\sigma.$$

To compute the forcing function, we only need $\Phi_{11}^\sigma(t, 0)$ above and to cut the integral in 4 different parts, of which only three can exist together. Informally, the integral on $\sigma$ is decomposed using Table 9.

Two regions correspond to the first case $\gamma_2 B \sigma \leq \sqrt{4\gamma_3 B}$ and the two other regions correspond to the second case $\gamma_2 B \sigma \geq \sqrt{4\gamma_3 B}$. The two sub-regions in each group correspond respectively to the sub-conditions $\sigma\sqrt{\gamma_3 B}(1+t) \gtrless 1$, $\gamma_2 B\sigma^2 t \gtrless 1$. Note that, as noted in Table 9, for $t \geq \frac{\gamma_2}{\gamma_3}$, the third regions disappears (and the lower-bound of the fourth integral becomes $\frac{\sqrt{4\gamma_3 B}}{\gamma_2 B}$). On the other hand, for $t \leq \frac{\gamma_2}{\gamma_3}$, the second integral disappears (and upper-bound of first integral becomes $\frac{\sqrt{4\gamma_3 B}}{\gamma_2 B}$).

**The case $t \geq \frac{\gamma_2}{\gamma_3}$**

**Proposition H.13.** *Let $\alpha > 0$, $2\alpha + 2\beta > 1$ and consider the general Parametrization H.1. Suppose that $t \geq \frac{\gamma_2}{\gamma_3}$. Then, if $\delta > \max\{2 + \frac{2\beta-1}{\alpha}, 2 - \frac{1}{\alpha}, 1\}$, $2\delta \notin \mathbb{N}$, for any $M > 0$, there exists a constant $C > 0$ such that*

$$\begin{cases} \mathcal{F}_{pp}(t) \asymp \left(\sqrt{\gamma_3 B}(1+t)\right)^{-2 - \frac{2\beta-1}{\alpha}} & \text{if} \quad \sqrt{\gamma_3 B}(1+t) \geq C \\ \mathcal{F}_{pp}(t) \asymp 1 & \text{if} \quad \sqrt{\gamma_3 B}(1+t) \leq C \end{cases}$$

*If additionally $\alpha > \frac{1}{2}$, $\beta > \frac{1}{2}$,*

$$\begin{cases} \mathcal{F}_{ac}(t) \asymp d^{-1} \left(\sqrt{\gamma_3 B}(1+t)\right)^{-2 + \frac{1}{\alpha}} & \text{if} \quad Md^\alpha \geq \sqrt{\gamma_3 B}(1+t) \geq C \\ \mathcal{F}_{ac}(t) \asymp d^{-1} & \text{if} \quad \sqrt{\gamma_3 B}(1+t) \leq C \end{cases}$$

*Proof.* We first consider the case $\frac{\sqrt{\gamma_3 B}}{\gamma_2 B} \lesssim 1$. For $t \geq \frac{\gamma_2}{\gamma_3}$, the third region disappears. Let $\epsilon = \frac{1}{2}$ and $\epsilon_1$ small enough, then we know the existence of some $c_\epsilon > 0$ small enough such that if $\frac{c_\epsilon}{\sqrt{\gamma_3 B}(1+t)} \leq (1-\epsilon_1)\frac{\sqrt{4\gamma_3 B}}{\gamma_2 B} \leq (1+\epsilon_1)\frac{\sqrt{4\gamma_3 B}}{\gamma_2 B} \leq 1$ we can decompose

$$\mathcal{F}_{pp}(t) \asymp \int_{\sigma=0}^{1} \sigma^{1+\frac{2\beta-1}{\alpha}} \Phi_{11}^{\sigma}(t,0) \, d\sigma$$

$$\asymp \int_{\sigma=0}^{\frac{c_\epsilon}{\sqrt{\gamma_3 B}(1+t)}} \sigma^{1+\frac{2\beta-1}{\alpha}} \Phi_{11}^{\sigma}(t,0) \, d\sigma + \int_{\sigma=\frac{c_\epsilon}{\sqrt{\gamma_3 B}(1+t)}}^{(1-\epsilon_1)\frac{\sqrt{4\gamma_3 B}}{\gamma_2 B}} \sigma^{1+\frac{2\beta-1}{\alpha}} \Phi_{11}^{\sigma}(t,0) \, d\sigma$$

$$+ \int_{\sigma=(1-\epsilon_1)\frac{\sqrt{4\gamma_3 B}}{\gamma_2 B}}^{(1+\epsilon_1)\frac{\sqrt{4\gamma_3 B}}{\gamma_2 B}} \sigma^{1+\frac{2\beta-1}{\alpha}} \Phi_{11}^{\sigma}(t,0) \, d\sigma + \int_{\sigma=(1+\epsilon_1)\frac{\sqrt{4\gamma_3 B}}{\gamma_2 B}}^{1} \sigma^{1+\frac{2\beta-1}{\alpha}} \Phi_{11}^{\sigma}(t,0) \, d\sigma$$

$$\asymp \int_{\sigma=0}^{\frac{c_\epsilon}{\sqrt{\gamma_3 B}(1+t)}} \sigma^{1+\frac{2\beta-1}{\alpha}} (1 \pm \epsilon) \, d\sigma$$

$$+ \int_{\sigma=\frac{c_\epsilon}{\sqrt{\gamma_3 B}(1+t)}}^{(1-\epsilon_1)\frac{\sqrt{4\gamma_3 B}}{\gamma_2 B}} \sigma^{1+\frac{2\beta-1}{\alpha}} \mathcal{O}^{+}(e^{-\gamma_2 B \sigma^2 t} \left(\sigma\sqrt{\gamma_3 B}(1+t)\right)^{-\delta}) \, d\sigma$$

$$+ \int_{\sigma=(1-\epsilon_1)\frac{\sqrt{4\gamma_3 B}}{\gamma_2 B}}^{(1+\epsilon_1)\frac{\sqrt{4\gamma_3 B}}{\gamma_2 B}} \sigma^{1+\frac{2\beta-1}{\alpha}} \mathcal{O}^{+}(e^{-\frac{\gamma_2 B \sigma^2 t}{2}}) \, d\sigma + \int_{\sigma=(1+\epsilon_1)\frac{\sqrt{4\gamma_3 B}}{\gamma_2 B}}^{1} \sigma^{1+\frac{2\beta-1}{\alpha}} \left(\gamma_2 B \sigma^2 (1+t)\right)^{-\delta} \, d\sigma$$

$$\asymp \left(\sqrt{\gamma_3 B}(1+t)\right)^{-2-\frac{2\beta-1}{\alpha}} + \mathcal{O}^{+}(\left(\sqrt{\gamma_3 B}(1+t)\right)^{-2-\frac{2\beta-1}{\alpha}} \int_{u=c_\epsilon}^{\frac{2\gamma_3(1+t)}{\gamma_2}} u^{1+\frac{2\beta-1}{\alpha}-\delta} \, du)$$

$$+ \mathcal{O}^{+}(\left(\frac{\sqrt{\gamma_3 B}}{\gamma_2 B}\right)^{2+\frac{2\beta-1}{\alpha}} e^{-\frac{\gamma_3}{\gamma_2}t})$$

$$+ \mathcal{O}^{+}(\left(\sqrt{\gamma_2 B}(1+t)\right)^{-2-\frac{2\beta-1}{\alpha}} e^{-\frac{\gamma_3}{\gamma_2}t} \int_{u=\sqrt{\frac{4\gamma_3(1+t)}{\gamma_2}}}^{\sqrt{\gamma_2 B(1+t)}} u^{-2\delta+1+\frac{2\beta-1}{\alpha}} \, du)$$

$$\asymp \left(\sqrt{\gamma_3 B}(1+t)\right)^{-2-\frac{2\beta-1}{\alpha}}$$

In the first integral we have used that $1+\frac{2\beta-1}{\alpha} > -1 \iff 2\alpha + 2\beta > 1$ and that $\Phi_{11}^{\sigma}(t,0) = 1 \pm \epsilon$ if $\sigma\sqrt{\gamma_3 B}(1+t) < c_\epsilon$. In the second integral we have made the change of variable $u = \sigma\sqrt{\gamma_3 B}(1+t)$ and used that $\delta > 2 + \frac{2\beta-1}{\alpha}$ for the integral on $u$ to converge when $\frac{\gamma_3}{\gamma_2}t \to \infty$. In the last integral, we have made the change of variable $u^2 = \gamma_2 B\sigma^2 t$. We use again that $\delta > 2 + \frac{2\beta-1}{\alpha}$ to get that it is negligible with respect to the other integrals

$$\left(\sqrt{\gamma_2 B(1+t)}\right)^{-2-\frac{2\beta-1}{\alpha}} \int_{u=\sqrt{\frac{4\gamma_3(1+t)}{\gamma_2}}}^{\sqrt{\gamma_2 B(1+t)}} u^{-2\delta+1+\frac{2\beta-1}{\alpha}} \, du$$

$$\lesssim \left(\sqrt{\gamma_2 B(1+t)}\right)^{-2-\frac{2\beta-1}{\alpha}} \sqrt{\frac{\gamma_3(1+t)}{\gamma_2}}^{-2-\frac{2\beta-1}{\alpha}}$$

$$\lesssim \left(\sqrt{\gamma_3 B}(1+t)\right)^{-2-\frac{2\beta-1}{\alpha}}$$

Additionally, note that for $\sigma\sqrt{\gamma_3 B}(1+t) \leq c_\epsilon$, we clearly have $\mathcal{F}_{pp}(t) \asymp \int_0^1 \sigma^{1+\frac{2\beta-1}{\alpha}}(1 \pm \epsilon) \, d\sigma \asymp 1$.

Finally, in the case $\frac{\sqrt{\gamma_3 B}}{\gamma_2 B} \gtrsim 1$, it is still clear that if $\sqrt{\gamma_3 B}(1+t) \lesssim 1$, only the first integral contributes and we obtain similarly $\mathcal{F}_{pp}(t) \asymp 1$ while if $\sqrt{\gamma_3 B}(1+t) \gtrsim 1$, the second integral is capped and we still obtain $\mathcal{F}_{pp}(t) \asymp (\sqrt{\gamma_3 B}(1+t))^{-2-\frac{2\beta-1}{\alpha}}$.

A similar calculation shows that $\mathcal{F}_{ac}(t) \asymp c_\beta d^{-1} \left(\sqrt{\gamma_3 B}(1+t)\right)^{-2+\frac{1}{\alpha}}$. The only difference is that we require $2 - \frac{1}{\alpha} > 0$ for the first integral to exist and hence to have $\alpha > \frac{1}{2}$. It's not a problem since below we have bounds from [80] (see Proposition H.15). Additionally, for the second integral to converge (and the third integral to be negligible) we require $\delta > 2 - \frac{1}{\alpha}$.

$$\square$$

**The case $t \leq \frac{\gamma_2}{\gamma_3}$**

**Proposition H.14.** *Suppose that $t \leq \frac{\gamma_2}{\gamma_3}$. Then, if $\delta > \max\{1 + \frac{2\beta - 1}{2\alpha}, 1 - \frac{1}{2\alpha}, 1\}$, $2\delta \notin \mathbb{N}$ and $2\alpha + 2\beta > 1$, for any $M > 0$ we have*

$$
\begin{cases}
\mathcal{F}_{pp}(t) \asymp \left(\sqrt{\gamma_2 B(1+t)}\right)^{-2 - \frac{2\beta - 1}{\alpha}} & \text{if} \quad \sqrt{\gamma_2 B(1+t)} \geq 1 \\
\mathcal{F}_{pp}(t) \asymp 1 & \text{if} \quad \sqrt{\gamma_2 B(1+t)} \leq 1
\end{cases}
$$

*If additionally $\alpha > \frac{1}{2}$ and $\beta > \frac{1}{2}$,*

$$
\begin{cases}
\mathcal{F}_{ac}(t) \asymp d^{-1} \left(\sqrt{\gamma_2 B(1+t)}\right)^{-2 + \frac{1}{\alpha}} & \text{if} \quad Md^{\alpha} \geq \sqrt{\gamma_2 B(1+t)} \geq 1 \\
\mathcal{F}_{ac}(t) \asymp d^{-1} & \text{if} \quad \sqrt{\gamma_2 B(1+t)} \leq 1
\end{cases}
$$

*Proof.* For $t \leq \frac{\gamma_2}{\gamma_3}$, the second region disappears. Let $\epsilon = \frac{1}{2}$, $\epsilon_1 > 0$ small enough and some $c_\epsilon > 0$. We first consider the case where $\frac{\sqrt{\gamma_3 B}}{\gamma_2 B} \lesssim 1$. For $(1 + \epsilon_1)\frac{\sqrt{4\gamma_3 B}}{\gamma_2 B} \leq \frac{c_\epsilon}{\sqrt{\gamma_2 B(1+t)}} \leq 1$, we write:

$$
\mathcal{F}_{pp}(t) \asymp \int_{\sigma = 0}^{(1 - \epsilon_1)\frac{\sqrt{4\gamma_3 B}}{\gamma_2 B}} \sigma^{1 + \frac{2\beta - 1}{\alpha}} (1 \pm \epsilon) \, d\sigma + \int_{(1 - \epsilon_1)\frac{\sqrt{4\gamma_3 B}}{\gamma_2 B}}^{(1 + \epsilon_1)\frac{\sqrt{4\gamma_3 B}}{\gamma_2 B}} \sigma^{1 + \frac{2\beta - 1}{\alpha}} \mathcal{O}^+(e^{-\frac{\gamma_2 B \sigma^2 t}{2}}) \, d\sigma
$$

$$
+ \int_{\sigma = \frac{\sqrt{(1 + \epsilon_1) 4\gamma_3 B}}{\gamma_2 B}}^{\frac{c_\epsilon}{\sqrt{\gamma_2 B(1+t)}}} \sigma^{1 + \frac{2\beta - 1}{\alpha}} (1 \pm \epsilon) \, d\sigma + \mathcal{O}^+\left(\int_{\sigma = \frac{c_\epsilon}{\sqrt{\gamma_2 B(1+t)}}}^{1} \sigma^{1 + \frac{2\beta - 1}{\alpha}} \left(\gamma_2 B \sigma^2 (1+t)\right)^{-\delta} \, d\sigma\right)
$$

$$
\asymp \left(\sqrt{\gamma_2 B(1+t)}\right)^{-2 - \frac{2\beta - 1}{\alpha}} + \mathcal{O}^+\left(\left(\frac{\sqrt{\gamma_3 B}}{\gamma_2 B}\right)^{2 + \frac{2\beta - 1}{\alpha}} e^{-\frac{\gamma_3}{\gamma_2} t}\right)
$$

$$
+ \mathcal{O}^+\left(\left(\sqrt{\gamma_2 B(1+t)}\right)^{-2 - \frac{2\beta - 1}{\alpha}} \int_{u = c_\epsilon}^{\sqrt{\gamma_2 B(1+t)}} u^{-2\delta + 1 + \frac{2\beta - 1}{\alpha}} \, du\right)
$$

$$
\asymp \left(\sqrt{\gamma_2 B(1+t)}\right)^{-2 - \frac{2\beta - 1}{\alpha}}
$$

Here we have used for both first and third integral that $1 + \frac{2\beta - 1}{\alpha} > -1 \iff 2\alpha + 2\beta > 1$. In the last integral, we have made the change of variable $u^2 = \gamma_2 B \sigma^2(1 + t)$ and used that $\delta > 1 + \frac{2\beta - 1}{2\alpha} \iff -2\delta + 1 + \frac{2\beta - 1}{\alpha} < -1$ for the integral on $u$ to converge. Also note that we bounded the contribution near the singular point by noticing that

$$
\frac{\sqrt{\gamma_3 B}}{\gamma_2 B} \leq \frac{1}{\sqrt{\gamma_2 B t}} \iff t \leq \frac{\gamma_2}{\gamma_3}.
$$

On the other hand, if $\sqrt{\gamma_2 B(1+t)} \leq c_\epsilon$, the last integral disappears, while the third one is cut at 1 and we obtain: $\mathcal{F}_{pp}(t) \asymp \int_0^1 \sigma^{1 + \frac{2\beta - 1}{\alpha}} \, d\sigma \asymp 1$.

Finally, in the case $\frac{\sqrt{\gamma_3 B}}{\gamma_2 B} \gtrsim 1$, we can check that necessarily $t \lesssim \frac{\gamma_2}{\gamma_3} \implies \sqrt{\gamma_2 B t} \asymp \frac{\gamma_2 B}{\sqrt{\gamma_3 B}} \lesssim 1$ while only the first integral remains and we obtain $\mathcal{F}_{pp}(t) \asymp 1$.

A similar calculation shows when $\alpha > \frac{1}{2}$, $\beta > \frac{1}{2}$ that $\mathcal{F}_{ac}(t) \asymp d^{-1} \left(\sqrt{\gamma_2 B(1+t)}\right)^{-2 + \frac{1}{\alpha}}$ if $\sqrt{\gamma_2 B(1+t)} > c_\epsilon$ and $\mathcal{F}_{ac}(t) \asymp d^{-1}$ if $\sqrt{\gamma_2 B(1+t)} \leq c_\epsilon$. In that case we need as previously that $\alpha > \frac{1}{2}$ in the first and third integral and $\delta > 1 - \frac{1}{2\alpha} \iff -2\delta + 1 - \frac{1}{\alpha} < -1$ for the last integral on $u$ to converge. $\square$

We can state two simple bounds on $\mathcal{F}_{ac}$.

**Proposition H.15.** *Let $\alpha > 0$, $2\alpha + 2\beta > 1$, $\alpha \neq \frac{1}{2}$, $\beta \neq \frac{1}{2}$. There exists a constant $C(\alpha, \beta)$ such that $\forall t \geq 0$:*

$$\begin{cases} \mathcal{F}_{ac}(t) \leq C \times \mathcal{F}_0(t) & \text{if} \quad 2\alpha < 1, 2\beta > 1 \\ \mathcal{F}_{ac}(t) = 0 & \text{if} \quad 2\beta < 1 \end{cases}$$

*Proof.* The proof is very similar to the one of [80, Proposition H.4]. First, if $2\beta > 1$, $c_\beta = 0$ and hence $\forall t \geq 0$, $\mathcal{F}_{ac}(t) = 0$.

Now, suppose that $2\alpha < 1, 2\beta > 1$. From Lemma H.1, we know that $\forall \sigma > 0, \forall t \geq 0$, $\Phi_{11}^\sigma(t, 0) \leq 1$. Hence this brings:

$$\begin{aligned} \mathcal{F}_{ac}(t) &\lesssim c_\beta \int_{d^{-\alpha}}^1 \sigma^{1 - \frac{1}{\alpha}} d^{-1} \, \mathrm{d}\sigma \\ &\lesssim c_\beta \left( \frac{d^{-2\alpha} - \frac{1}{d}}{\frac{1}{\alpha} - 2} \right) \\ &\lesssim d^{-2\alpha} \end{aligned}$$

Since we know that $\mathcal{F}_0(t) \asymp d^{-2\alpha + \max\{0, 1 - 2\beta\}}$, we get the result.

$\square$

**Corollary H.1.** $\forall M > 0$, $\exists C > 0$, $\forall t > 0$, *if* $\max\{\gamma_2 B(1 + t), (\sqrt{\gamma_3 B}(1 + t)^2\} > M d^{2\alpha}$ *we have:*

$$\mathcal{F}_{pp}(t) + \mathcal{F}_{ac}(t) \leq C \mathcal{F}_0(t)$$

*Proof.* This is a consequence of Propositions H.12 to H.15. $\square$

Finally, we relate the forcing function $\mathcal{F}(t)$ to the terms $\mathcal{F}_0(t), \mathcal{F}_{pp}(t), \mathcal{F}_{ac}(t)$ that we just estimated.

**Proposition H.16.** *Let $\alpha, \beta > 0$, $2\alpha + 2\beta > 1$, $\alpha + 1 > \beta$, $\alpha, \beta \neq \frac{1}{2}$. Use Parametrization H.1 with $\delta > \max\{1, 2 + \frac{2\beta - 1}{\alpha}, 2 - \frac{1}{\alpha}\}$ and $2\delta \notin \mathbb{N}$. Then there exists a constant $C(\alpha, \beta, H)$ such that $\forall t \geq 0$,*

$$\frac{1}{C} \left( \mathcal{F}_0(t) + \mathcal{F}_{ac}(t) + \mathcal{F}_{pp}(t) \right) \leq \mathcal{F}(t) \leq C \left( \mathcal{F}_0(t) + \mathcal{F}_{ac}(t) + \mathcal{F}_{pp}(t) \right).$$

*Proof.* It is clear that for any $t \geq 0$ and on the range of $\sigma$'s where we can apply Proposition H.5 and Proposition H.8 that $\Phi_{11}^\sigma(t, 0) \asymp 1$. More precisely it corresponds to the range $\sigma \lesssim \frac{1}{\sqrt{\gamma_3 B t}}$ and $\gamma_2 B \sigma \lesssim \sqrt{4\gamma_3 B}$ or the range $\sigma \lesssim \frac{1}{\sqrt{\gamma_2 B t}}$ and $\gamma_2 B \sigma \gtrsim \sqrt{4\gamma_3 B}$. Hence the function $\left( \sigma \mapsto \Phi_{11}^\sigma(t, 0) \right)$ satisfies, up to a constant, the hypothesis of Proposition D.13 on this range of $\sigma$'s. The reader can check that the contribution from eigenvalues of this range lower-bounds $\mathcal{F}_{pp}(t), \mathcal{F}_{ac}(t)$.

Similarly, by upper-bounding the oscillatory part in cosine/sine appearing in Proposition H.9 and Proposition H.6 by constants, one can upper-bound $\Phi_{11}^\sigma(t, s)$ by some function which satisfies the hypothesis in Proposition D.13.

This implies the bound for some $M, M_1, M_2 > 0$ and some $C > 0$

$$\begin{aligned} \frac{1}{C} \int_{M_1 d^{-2\alpha}}^{\frac{1}{M_1}} \Phi_{11}^\sigma(t, 0)(\mu_{\mathcal{F}_{pp}} + \mu_{\mathcal{F}_{ac}})(\mathrm{d}\sigma^2) &\leq \int_{M d^{-2\alpha}}^{\frac{1}{M}} \Phi_{11}^\sigma(t, 0)\mu_{\mathcal{F}}(\mathrm{d}\sigma^2) \\ &\leq C \int_{M_2 d^{-2\alpha}}^{\frac{1}{M_2}} \Phi_{11}^\sigma(t, 0)(\mu_{\mathcal{F}_{pp}} + \mu_{\mathcal{F}_{ac}})(\mathrm{d}\sigma^2). \end{aligned}$$

There remains to bound the integrals on segments $[0, M d^{2\alpha}] \cup [\frac{1}{M}, \infty]$ for some constants $M > 0$. We know from Lemma H.1 that for any $t \geq 0$ and any

| DANA-constant with $t \leq 2\frac{\gamma_2}{\gamma_3}$ | DANA-constant with $t \geq 2\frac{\gamma_2}{\gamma_3}$ |
|---|---|
| $\mathcal{F}_0(t) \asymp d^{-2\alpha+\max\{0,1-2\beta\}}$ | $\mathcal{F}_0(t) \asymp d^{-2\alpha+\max\{0,1-2\beta\}}$ |
| $\mathcal{F}_{pp}(t) \asymp (\gamma_2 Bt)^{-1-\frac{2\beta-1}{2\alpha}}$ | $\mathcal{F}_{pp}(t) \asymp \left(\sqrt{\gamma_3 B}\cdot t\right)^{-2-\frac{2\beta-1}{\alpha}}$ |
| $\mathcal{F}_{ac}(t) \leq \begin{cases} C \times \mathcal{F}_0(t), & \text{if } 2\beta > 1, 2\alpha < 1 \\ 0, & \text{if } 2\beta < 1 \end{cases}$ 
 if $2\beta > 1, 2\alpha > 1$, $\mathcal{F}_{ac}(t) \asymp (B\gamma_2 t)^{-1+\frac{1}{2\alpha}} d^{-1}$ | $\mathcal{F}_{ac}(t) \asymp \begin{cases} C \times \mathcal{F}_0(t), & \text{if } 2\beta > 1, 2\alpha > 1 \\ 0, & \text{if } 2\beta < 1. \end{cases}$ 
 if $2\beta > 1, 2\alpha > 1$, $\mathcal{F}_{ac}(t) \asymp \left(\sqrt{\gamma_3 B}\cdot t\right)^{-2+\frac{1}{\alpha}} d^{-1}$ |
| $\mathcal{K}_{pp}(t,0) \asymp B\gamma_2^2 (\gamma_2 Bt)^{-2+1/(2\alpha)}$ | $\mathcal{K}_{pp}(t,0) \asymp B\gamma_2^2 \left(\sqrt{\gamma_3 B}\cdot t\right)^{-4+1/\alpha}$ |

Table 10: **Large $d$ behavior of the forcing function and kernel function for DANA-constant for small and large $t$.** Here the constant $C$ is independent of dimension.

$\sigma > 0$, $\Phi_{11}^\sigma(t,0) \leq 1$. Hence using Propositions D.4, D.6 and H.12 we can bound for any $M > 0$, $\max\{\int_{0+}^{Md^{-2\alpha}} \Phi_{11}^\sigma(t,0)\mu_{\mathcal{F}}(d\sigma^2), \int_{0+}^{Md^{-2\alpha}} \Phi_{11}^\sigma(t,0)(\mu_{\mathcal{F}_{pp}} + \mu_{\mathcal{F}_{ac}})(d\sigma^2)\} \lesssim d^{-2\alpha+(1-2\beta)_+} \lesssim \mathcal{F}_0(t)$. Additionally, using Propositions D.2 and H.9, we bound $\max\{\int_{\frac{1}{M}}^{\infty} \Phi_{11}^\sigma(t,0)\mu_{\mathcal{F}}(d\sigma^2), \int_{\frac{1}{M}}^{\infty} \Phi_{11}^\sigma(t,0)(\mu_{\mathcal{F}_{pp}} + \mu_{\mathcal{F}_{ac}})(d\sigma^2)\} \lesssim \min\{e^{-\gamma_2 Bt/(2M)}, (\sqrt{\gamma_3 B/M}t)^{-\delta}\} \ll \mathcal{F}_0(t) + \mathcal{F}_{ac}(t) + \mathcal{F}_{pp}(t)$. This concludes the proof. $\qquad\square$

### H.8 Necessary conditions for stability

In this section, we look for *necessary conditions* on learning rates and batch exponents $\kappa_1, \kappa_2, \kappa_b > 0$ in Parametrization H.1 for the risk to remain bounded. We first state a technical lemma that shows divergence of the solution to a Volterra equation when the forcing function is lower-bounded and the kernel noise is too large.

**Lemma H.3.** *Let $\mathcal{F} : \mathbb{R}_+ \to \mathbb{R}_+$, $\mathcal{K} : \{(t,s) \in \mathbb{R}_+^2, t \geq s\} \to \mathbb{R}_+$. Let $\mathcal{P} : \mathbb{R}_+ \to \mathbb{R}_+$ solution to the Volterra equation $\mathcal{P}(t) = \mathcal{F}(t) + \int_0^t \mathcal{P}(s)\mathcal{K}(t,s)\,\mathrm{d}s$. Suppose that:*

- $\forall t \geq 0$, $\mathcal{F}(t) \geq \mathcal{F}_0 > 0$,

- $\liminf_{t\geq 0} \int_{t/2}^t \mathcal{K}(t,s)\,\mathrm{d}s > 1$.

*Then,*
$$\lim_{t\to\infty} \mathcal{P}(t) = \infty$$

*Proof.* By assumption, we know the existence of $T > 0$ such that $\forall t \geq T$, $\int_{t/2}^t \mathcal{K}(t,s)\,\mathrm{d}s > 1 + \epsilon$ for some $\epsilon > 0$. It is then clear, by recursion, that
$$\forall k \in \mathbb{N} \cup \{-1\}, \ \forall t \geq T \times 2^k, \quad \mathcal{P}(t) \geq (1+\epsilon)^{k+1}\mathcal{F}_0.$$
This proves that $\mathcal{P}(t) \overset{t\to\infty}{\to} \infty$. $\qquad\square$

**Lemma H.4.** *Let $\alpha > \frac{1}{4}$. Let $\delta > \max\{1, 4 - \frac{1}{\alpha}\}$, $2\delta \notin \mathbb{N}$. Under Parametrization H.1, if $\kappa_1 < (1-2\alpha)_+$ or $\kappa_2 < \kappa_1 + 1 - \kappa_b$, then for $d$ large enough,*
$$\liminf_{t\geq 0} \int_{t/2}^t \mathcal{K}(t,s)\,\mathrm{d}s > 1.$$

*Proof.* We first consider the case where $\kappa_2 < 2\kappa_1 + 2\alpha - \kappa_b$. This ensures that $\frac{\sqrt{\gamma_3 B}}{\gamma_2 B} \gtrsim d^{-\alpha}$.

Since $\forall t \geq s \geq 0$, $\forall \sigma \geq 0$, $\Phi_{11}^\sigma(t,s), \Phi_{12}^\sigma(t,s) \geq 0$, we use Fubini theorem to write for $t$ large enough so that $\frac{1}{\sqrt{\gamma_3 B}(1+t/2)} \leq \frac{\sqrt{\gamma_3 B}}{\gamma_2 B}$

$$\int_{t/2}^{t} \mathcal{K}(t,s)\,\mathrm{d}s \geq \gamma_2^2 B \int_{\sigma=\frac{1}{\sqrt{\gamma_3 Bs}}}^{\frac{\sqrt{\gamma_3 B}}{\gamma_2 B}} \left( \int_{t/2}^{t} \Phi_{11}^{\sigma}(t,s)\,\mathrm{d}s \right) \mu_{\mathcal{K}}(\mathrm{d}\sigma^2)$$

$$+ B \int_{\sigma=\frac{1}{\sqrt{\gamma_3 Bs}}}^{\frac{\sqrt{\gamma_3 B}}{\gamma_2 B}} \left( \int_{t/2}^{t} \Phi_{12}^{\sigma}(t,s)\,\mathrm{d}s \right) \mu_{\mathcal{K}}(\mathrm{d}\sigma^2).$$

Note that we integrate only up to $\frac{\sqrt{\gamma_3 B}}{\gamma_2 B} < \frac{\sqrt{4\gamma_3 B}}{\gamma_2 B}$ as we only look for a lower-bound on the kernel and this ensures we remain bounded away from the singular point. For the first term, we write for $\sigma \in \left[ \frac{1}{\sqrt{\gamma_3 B}(1+s)}, \frac{\sqrt{\gamma_3 B}}{\gamma_2 B} \right]$,

$$\int_{t/2}^{t} \Phi_{11}^{\sigma}(t,s)\,\mathrm{d}s = \int_{t/2}^{t} \frac{1}{2} e^{-\gamma_2 B\sigma^2(t-s)} \left( \frac{1+t}{1+s} \right)^{-\delta} \left( 1 + \cos(\sigma\sqrt{4\gamma_3 B}(t-s)) \right.$$

$$\left. + \log\left( \frac{1+t}{1+s} \right) \frac{\delta\gamma_2 B\sigma}{\sqrt{4\gamma_3 B - \gamma_2^2 B^2\sigma^2}} \right) + \left( \mathcal{O}\left( \frac{\gamma_2 B\sigma}{\sqrt{\gamma_3 B}} \right) + \mathcal{O}(\epsilon) \right) \right) \mathrm{d}s$$

$$\gtrsim \left( \mathcal{O}\left( \frac{\gamma_2 B\sigma}{\sqrt{\gamma_3 B}} \right) + \mathcal{O}(\epsilon) \right) \times \frac{1}{\gamma_2 B\sigma^2} + \frac{1}{\gamma_2 B\sigma^2}$$

$$\gtrsim \frac{1}{\gamma_2 B\sigma^2}.$$

Here we used that $\sigma\sqrt{\gamma_3 B} \geq \gamma_2 B\sigma^2$ and $t$ large enough.

For the second term, a similar argument brings that again for $\sigma \in \left[ \frac{1}{\sqrt{\gamma_3 B}(1+s)}, \frac{\sqrt{\gamma_3 B}}{\gamma_2 B} \right]$,

$$\int_{t/2}^{t} \Phi_{12}^{\sigma}(t,s)\,\mathrm{d}s = \int_{t/2}^{t} \frac{\gamma_3}{2B\sigma^2} e^{-\gamma_2 B\sigma^2(t-s)} \left( \frac{1+t}{1+s} \right)^{-\delta} \left( 1 - \cos(\sigma\sqrt{4\gamma_3 B}(t-s)) \right.$$

$$\left. + \log\left( \frac{1+t}{1+s} \right) \frac{\delta\gamma_2 B\sigma}{\sqrt{4\gamma_3 B - \gamma_2^2 B^2\sigma^2}} \right) + \left( \mathcal{O}\left( \frac{\gamma_2 B\sigma}{\sqrt{\gamma_3 B}} \right) + \mathcal{O}(\epsilon) \right) \right) \mathrm{d}s$$

$$\gtrsim \frac{\gamma_3}{B\sigma^2} \left( \left( \mathcal{O}\left( \frac{\gamma_2 B\sigma}{\sqrt{\gamma_3 B}} \right) + \mathcal{O}(\epsilon) \right) \times \frac{1}{\gamma_2 B\sigma^2} + \frac{1}{\gamma_2 B\sigma^2} \right)$$

$$\gtrsim \frac{\gamma_3}{B\sigma^2} \left( \frac{1}{\gamma_2 B\sigma^2} \right).$$

Here we used that $\gamma_2 B\sigma^2 < \sigma\sqrt{4\gamma_3 B}$ to obtain that the oscillations from $\cos$ average in the integral.

It is hence clear using that $\frac{\sqrt{\gamma_3 B}}{\gamma_2 B} \gtrsim d^{-\alpha}$ and $t$ large enough that

$$\int_{t/2}^{t} \mathcal{K}(t,s)\,\mathrm{d}s \gtrsim \gamma_2^2 B \int_{\sigma=\frac{1}{\sqrt{\gamma_3 B}(1+s)}}^{\frac{\sqrt{\gamma_3 B}}{\gamma_2 B}} \frac{1}{\gamma_2 B\sigma^2} \mu_{\mathcal{K}}(\mathrm{d}\sigma^2) + B \int_{\sigma=\frac{1}{\sqrt{\gamma_3 B}(1+s)}}^{\frac{\sqrt{\gamma_3 B}}{\gamma_2 B}} \frac{\gamma_3}{B\sigma^2} \times \frac{1}{\gamma_2 B\sigma^2} \mu_{\mathcal{K}}(\mathrm{d}\sigma^2)$$

$$\gtrsim \int_{d^{-\alpha}}^{\min\{1, \frac{\sqrt{\gamma_3 B}}{\gamma_2 B}\}} \left( \gamma_2^2 B\sigma^{3-1/\alpha} \frac{1}{\gamma_2 B\sigma^2} + B\sigma^{3-1/\alpha} \frac{\gamma_3}{\gamma_2 B^2\sigma^4} \right) \mathrm{d}\sigma$$

$$\gtrsim \gamma_2 d^{(1-2\alpha)_+} + \frac{\gamma_3}{\gamma_2 B} \times d.$$

The above shows that supposing $\kappa_2 < 2\kappa_1 + 2\alpha - \kappa_b$, then if $\kappa_1 < (1-2\alpha)_+$ or $\kappa_2 < \kappa_1 + 1 - \kappa_b$, then

$$\liminf_{t\to\infty} \int_{t/2}^{t} \mathcal{K}(t,s)\,\mathrm{d}s = +\infty.$$

We know consider the second case where $\kappa_2 \geq 2\kappa_1 + 2\alpha - \kappa_b$. It is clear that we only need to consider the case where $\kappa_1 < (1 - 2\alpha)_+$ since

$$\begin{cases} \kappa_1 \geq (1 - 2\alpha)_+ \\ \text{and } \kappa_2 \geq 2\kappa_1 + 2\alpha - \kappa_b \end{cases} \implies \kappa_2 \geq \kappa_1 + 1 - \kappa_b.$$

We use Proposition H.10 and obtain for given $M > 0$, $\sigma > \max\{\frac{\sqrt{\gamma_3 B}}{\gamma_2 B}, Md^{-\alpha}\}$ and $\gamma_2\sigma^2(1+s) > 1$ that

$$\Phi_{11}(t,s) = (1 + \mathcal{O}((\gamma_2 B\sigma^2(1+s))^{-1} + x)e^{\mu_1(t-s)} \left(\frac{1+t}{1+s}\right)^{\delta_1}$$

$$+ \mathcal{O}(x^4 + x^2(\gamma_2 B\sigma^2(1+s))^{-2} + (\gamma_2 B\sigma^2(1+t))^{-2})e^{\mu_2(t-s)} \left(\frac{1+t}{1+s}\right)^{\delta_2}$$

$$+ \mathcal{O}(x^2 + x(\gamma_2 B\sigma^2(1+s))^{-1} + (\gamma_2 B\sigma^2(1+t))^{-2})e^{\mu_3(t-s)} \left(\frac{1+t}{1+s}\right)^{\delta_3}.$$

Especially, the most important term is $e^{\mu_1(t-s)}$ which gives rise to $\frac{1}{2\gamma_2 B\sigma^2 t}$ in the kernel norm. To see that, notice that we can take $s$ large enough so that $(\gamma_2 B\sigma^2(1+s))^{-1}$ is negligible and $x^2 = \frac{\gamma_3 B}{\gamma_2^2\sigma^2 B^2}$ as small as we want by increasing $d$ (or $M$ in the particular case where $\kappa_2 = 2\kappa_1 + 2\alpha$). This brings

$$\int_{t/2}^t (1 + \mathcal{O}((\gamma_2 B\sigma^2(1+s))^{-1} + x^2)e^{\mu_1(t-s)} \left(\frac{1+t}{1+s}\right)^{\delta_1} \mathrm{d}s \gtrsim \frac{1}{\mu_1} \gtrsim \frac{1}{\gamma_2 B\sigma^2}.$$

It is additionally clear that the last term brings a negligible contribution since

$$\int_{t/2}^t \mathcal{O}(x^2 + (\gamma_2 B\sigma^2(1+s))^{-1})e^{\mu_3(t-s)} \left(\frac{1+t}{1+s}\right)^{\delta_3} = \mathcal{O}(x^2 + (\gamma_2 B\sigma^2(1+s))^{-1}) \times \frac{1}{\mu_3}$$

$$= \mathcal{O}(x^2 + (\gamma_2 B\sigma^2(1+s))^{-1}) \times \frac{1}{\mu_1}.$$

We mostly need to bound the second term which is done as follows

$$\int_{t/2}^t \mathcal{O}(x^4 + (\gamma_2 B\sigma^2(1+s))^{-2})e^{\mu_2(t-s)} \left(\frac{1+t}{1+s}\right)^{\delta_2} \mathrm{d}s = \mathcal{O}((x^4 + (\gamma_2 B\sigma^2(1+t))^{-2})\frac{1}{\mu_2})$$

$$= \mathcal{O}\left(\left(\frac{\sqrt{\gamma_3 B}}{\gamma_2 B\sigma}\right)^4 \times \frac{\gamma_2}{\gamma_3}\right)$$

$$= \mathcal{O}\left(\left(\frac{\sqrt{\gamma_3 B}}{\gamma_2 B\sigma}\right)^2 \times \frac{1}{\gamma_2 B\sigma^2}\right).$$

We noticed that $\mu_2 \asymp \frac{\gamma_3}{\gamma_2}$ and $x = \frac{\sqrt{\gamma_3 B}}{\gamma_2 B\sigma}$. Additionally, we noticed that we can take $t$ as large as we want. Additionally, $\left(\frac{\sqrt{\gamma_3 B}}{\gamma_2 B\sigma}\right)^2 \times \frac{\gamma_2}{\gamma_3} = \frac{1}{\gamma_2 B\sigma^2}$.

Finally, we obtain that

$$\int_{t/2}^t \mathcal{K}(t,s)\,\mathrm{d}s \gtrsim \int_{\sigma=d^{-\alpha}}^1 \gamma_2^2 B \frac{1}{\gamma_2 B\sigma^2}\mu_{\mathcal{K}}(\mathrm{d}\sigma^2)$$

$$\gtrsim \int_{\sigma=d^{-\alpha}}^1 \gamma_2 \sigma^{1-1/\alpha)}\,\mathrm{d}\sigma$$

$$\gtrsim \gamma_2 d^{(1-2\alpha)+}.$$

The above hence implies that supposing $\kappa_2 \geq 2\kappa_1 + 2\alpha - \kappa_b$, then if $\kappa_1 < (2\alpha - 1)_+$ we have

$$\liminf \int_{t/2}^t \mathcal{K}(t,s)\,\mathrm{d}s = +\infty.$$

$\square$

**Corollary H.2** (Necessary condition for stability of DANA-constant). *Let $\alpha > \frac{1}{4}$. Let $\delta > \max\{1, 4 - \frac{1}{\alpha}\}$, $2\delta \notin \mathbb{N}$. Under Parametrization H.1, suppose $\kappa_1 < (1 - 2\alpha)_+$ or $\kappa_2 < \kappa_1 + 1 - \kappa_b$. Then for $d$ large enough, $\mathcal{P}(t) \overset{t \to \infty}{\to} \infty$.*

*Proof.* This is a direct consequence of Lemma H.3 and Lemma H.4 $\square$

## H.9   Sufficient condition for stability: upper-bound on the kernel norm

**Lemma H.5** (Sufficient condition for stability of DANA-constant). *Under Parametrization H.1, for given $\alpha > \frac{1}{4}, \alpha \neq \frac{1}{2}, \delta > \max\{1, 4 - \frac{1}{\alpha}\}$, $2\delta \notin \mathbb{N}$, $M > 0$, we know the existence of $C > 0$ such that,*

$$\forall t \geq 0, \ \int_0^t \mathcal{K}(t,s)\,\mathrm{d}s \leq C \left( \gamma_2 d^{\min\{0, 2\alpha - 1\}} + d \frac{\gamma_3}{\gamma_2 B} \right).$$

*As a consequence, there exists some $c > 0$ such that for any $\kappa_1 \geq (1 - 2\alpha)_+$, $\kappa_2 \geq \kappa_1 - \kappa_b + 1$, $\tilde{\gamma}_2 \leq c, \frac{\tilde{\gamma}_3}{\tilde{\gamma}_2 \times c_b} \leq c$ and any $d \geq 1$ we know that*

$$\sup_{t \geq 0} \mathcal{P}(t) < \infty.$$

**Remark H.2.** *We believe the following stronger bounds to be true and could be shown in the same way, although a little bit more technical. We discuss this in more detail in Section J. For given $\alpha > \frac{1}{4}, 2\alpha + 2\beta > 1$, $\alpha, \beta \neq \frac{1}{2}$, using Parametrization H.1 with parametrization vector $H$, $\delta > \max\{1, 4 - \frac{1}{\alpha}\}$, $2\delta \notin \mathbb{N}$ and for any $M > 0$ there exists $C(\alpha, \beta, H, M) > 0$ such that for any $t \geq 0$ with $\frac{1}{M} \leq \gamma_2 Bt \leq Md^{2\alpha}$,*

$$\frac{1}{C} \left( \gamma_2 (\gamma_2 Bt)^{\frac{(1-2\alpha)_+}{2\alpha}} + \frac{\gamma_3}{\gamma_2 B}(\gamma_2 Bt)^{1/(2\alpha)} \right) \leq \int_0^t \mathcal{K}(t,s)\,\mathrm{d}s$$

$$\leq C \left( \gamma_2 (\gamma_2 Bt)^{\frac{(1-2\alpha)_+}{2\alpha}} + \frac{\gamma_3}{\gamma_2 B}(\gamma_2 Bt)^{1/(2\alpha)} \right).$$

*Additionally the kernel norm converges in the sense that $\exists C(\alpha, \beta, H, M) > 0$ such that for any $t \geq 0$ with $\frac{1}{M} \leq \gamma_2 Bt \geq Md^{2\alpha}$*

$$\frac{1}{C} \left( \gamma_2 d^{(1-2\alpha)_+} + d\frac{\gamma_3}{\gamma_2 B} \right) \leq \int_0^t \mathcal{K}(t,s)\,\mathrm{d}s \leq C \left( \gamma_2 d^{(1-2\alpha)_+} + d\frac{\gamma_3}{\gamma_2 B} \right).$$

*Proof.* We can bound $\Phi_{11}(t,s), \Phi_{12}(t,s)$ as follows

- from Propositions H.5 to H.7 if $\gamma_2 \sigma B \leq (1 - \epsilon)\sqrt{\gamma_3 B}$, then $\Phi_{11}(t,s) = \mathcal{O}(e^{-\gamma_2 \sigma^2 B(t-s)})$, $\Phi_{12}(t,s) = \frac{\gamma_3}{B\sigma^2}\mathcal{O}(e^{-\gamma_2 \sigma^2 B(t-s)})$ with a corresponding lower-bound,

- for $\gamma_2 B\sigma \in [(1 - \epsilon)\sqrt{\gamma_3 B}, (1 + \epsilon)\sqrt{\gamma_3 B}]$, Proposition H.11 brings that for $t, s \geq 0$, $\Phi_{11}(t,s) = \mathcal{O}(e^{-c_\epsilon \gamma_2 \sigma^2 B(t-s)})$, $\Phi_{12}(t,s) = \frac{\sqrt{\gamma_3}}{\gamma_2 B\sigma^2}\mathcal{O}(e^{-c_\epsilon \gamma_2 \sigma^2 B(t-s)})$.

- Propositions H.8 and H.9 bring that for $\gamma_2 \sigma B \geq (1 + \epsilon)\sqrt{\gamma_3 B}$ if $\gamma_2 B\sigma^2 s \leq 1$, then $\Phi_{11}(t,s) = \mathcal{O}(e^{-c_\epsilon \gamma_2 \sigma^2 B(t-s)} + (\gamma_2 \sigma^2 B(t - s))^{-C_\epsilon})$, $\Phi_{12}(t,s) = \frac{\gamma_3}{B\sigma^2}\mathcal{O}(e^{-c_\epsilon \gamma_2 \sigma^2 B(t-s)} + (\gamma_2 \sigma^2 B(t - s))^{-C_\epsilon})$ with $c_\epsilon > 0, C_\epsilon > 2\delta > 2$. Finally, Proposition H.10 shows that $\Phi_{11}(t,s) = \mathcal{O}(e^{-c_\epsilon \gamma_2 \sigma^2 B(t-s)} + ((\gamma_2 \sigma^2 Bt)^{-2} +$

$$x^2)e^{-\frac{(t-s)\gamma_3}{\gamma_2}}), \ \Phi_{12}(t,s) = \frac{\gamma_3}{B\sigma^2}\mathcal{O}(e^{-c_\epsilon\gamma_2\sigma^2 B(t-s)} + ((\gamma_2\sigma^2 Bt)^{-2} + x^2)e^{-\frac{(t-s)\gamma_3}{\gamma_2}}) \text{ with}$$
$$x = \frac{\sqrt{\gamma_3 B}}{\gamma_2 B\sigma}.$$

We just need to integrate these estimates on $s$ and $\sigma$. It is clear that for $\sigma \leq (1+\epsilon)\frac{\sqrt{\gamma_3 B}}{\gamma_2 B}$ we have

$$\int_{s=0}^t \Phi_{11}(t,s)\,\mathrm{d}s \lesssim \int_{s=0}^t e^{-c_\epsilon\gamma_2 B\sigma^2(t-s)}\,\mathrm{d}s \lesssim \frac{1}{\gamma_2 B\sigma^2}$$

and similarly $\quad \int_{s=0}^t \Phi_{12}(t,s)\,\mathrm{d}s \lesssim \int_{s=0}^t \frac{\gamma_3}{B\sigma^2}e^{-c_\epsilon\gamma_2 B\sigma^2(t-s)}\,\mathrm{d}s \lesssim \frac{\gamma_3}{B\sigma^2} \times \frac{1}{\gamma_2 B\sigma^2}$

On the other hand, for $\sigma \geq (1+\epsilon)\frac{\sqrt{\gamma_3 B}}{\gamma_2 B}$, we still obtain

$$\int_{s=0}^t \Phi_{11}(t,s)\,\mathrm{d}s \lesssim \int_{s=0}^t \left(e^{-c_\epsilon\gamma_2 B\sigma^2(t-s)} + \left(\left(\frac{\sqrt{\gamma_3 B}}{\gamma_2 B\sigma}\right)^2 + \frac{1}{(\gamma_2 B\sigma^2 t)^2}\right)e^{-(t-s)\frac{\gamma_3}{\gamma_2}}\right)\mathrm{d}s$$
$$\lesssim \frac{1}{\gamma_2 B\sigma^2}.$$

Here we used that $\int_0^t e^{-(t-s)\frac{\gamma_3}{\gamma_2}}\,\mathrm{d}s \lesssim \min\{t, \frac{\gamma_2}{\gamma_3}\}$. We combined that with

$$\left(\frac{\sqrt{\gamma_3 B}}{\gamma_2 B\sigma}\right)^2 \frac{\gamma_2}{\gamma_3} \lesssim \frac{1}{\gamma_2 B\sigma^2} \quad \frac{1}{(\gamma_2 B\sigma^2 t)^2}t \lesssim \frac{1}{\gamma_2 B\sigma^2}\frac{1}{\gamma_2 B\sigma^2 t} \lesssim \frac{1}{\gamma_2 B\sigma^2}$$

where we used that in that range, $\frac{1}{\gamma_2 B\sigma^2 t} \lesssim 1$. Similarly, we have

$$\int_{s=0}^t \Phi_{12}(t,s)\,\mathrm{d}s \lesssim \frac{\gamma_3}{B\sigma^2} \times \frac{1}{\gamma_2 B\sigma^2}.$$

Now integrating against $\mu_{\mathcal{K}}$ and using Proposition D.14 brings the desired upper-bound for some constants $c, C > 0$,

$$\int_{s=0}^t \mathcal{K}(t,s)\,\mathrm{d}s = \int_0^t (\gamma_2^2 B\Phi_{11}(t,s) + B\Phi_{12}(t,s))\,\mathrm{d}s$$
$$\lesssim \int_{cd^{-2\alpha}}^1 \sigma^{3-1/\alpha}\left(\gamma_2^2 B\frac{1}{\gamma_2 B\sigma^2} + B\frac{\gamma_3}{B\sigma^2} \times \frac{1}{\gamma_2 B\sigma^2}\right)\mathrm{d}\sigma$$
$$\leq C\left(\gamma_2 \times d^{(1-2\alpha)_+} + \frac{\gamma_3}{\gamma_2 B} \times d\right).$$

The last claim is a straightforward consequence using the fact that $\mathcal{F}(t)$ is bounded from Proposition H.16 and $\mathcal{K}(t,s)$ is continuous. $\qquad\square$

## H.10  Upper-bound on the kernel function

In all this section, we assume $\kappa_1 = (1-2\alpha)_+$, $0 \leq \kappa_b \leq \kappa_1$, $\kappa_2 = \kappa_1 - \kappa_b + 1$. The previous sections show that these are the largest stable learning rates. In this section we will estimate an upper-bound on the pure-point term of the kernel function by:

- upper-bounding $|\cos(t)|, |\sin(t)| \leq 1$ in Propositions H.7 and H.10, hence defining $\bar{\Phi}_{11}, \bar{\Phi}_{12}$,
- using the upper bound on $\mu_{\mathcal{K}}$ by $\mu_{\mathcal{K}_{pp}}$ from in Section D.

| $\sigma \leq \dfrac{\sqrt{4\gamma_3 B}}{\gamma_2 B}$ | | |
|---|---|---|
| $\sigma < \dfrac{1}{\sqrt{\gamma_3 B}(1+t)}$ | $\dfrac{1}{\sqrt{\gamma_3 B}(1+t)} < \sigma < \dfrac{1}{\sqrt{\gamma_3 B}(1+s)}$ $(\implies 1+t > \frac{\gamma_2}{2\gamma_3})$ | $\dfrac{1}{\sqrt{\gamma_3 B}(1+s)} < \sigma$ $(\implies 1+s > \frac{\gamma_2}{2\gamma_3})$ |
| $\sigma \geq \dfrac{\sqrt{4\gamma_3 B}}{\gamma_2 B}$ | | |
| $\sigma < \dfrac{1}{\sqrt{\gamma_2 B t}}$ $(\implies t < \frac{\gamma_2}{4\gamma_3})$ | $\dfrac{1}{\sqrt{\gamma_2 B t}} < \sigma < \dfrac{1}{\sqrt{\gamma_2 B s}}$ $(\implies s < \frac{\gamma_2}{4\gamma_3})$ | $\dfrac{1}{\sqrt{\gamma_2 B s}} < \sigma$ |

Table 11: Cuts of $\sigma$ domains for kernel function

We hence define

$$\bar{\mathcal{K}}(t,s) \overset{\text{def}}{=} \bar{\mathcal{K}}^1(t,s) + \bar{\mathcal{K}}^2(t,s)$$

$$\overset{\text{def}}{=} \gamma_2^2 B \int_{\sigma=0}^{\infty} \bar{\Phi}_{11}^{\sigma}(t,s)\,\mathrm{d}\mu(\sigma^2) + \gamma_1^2 B \int_{\sigma=0}^{1} \bar{\Phi}_{12}^{\sigma}(t,s)\,\mathrm{d}\mu(\sigma^2).$$

We additionally define the corresponding pure-point contribution of the upper-bound on the kernel.

$$\bar{\mathcal{K}}_{pp}(t,s) \overset{\text{def}}{=} \bar{\mathcal{K}}_{pp}^1(t,s) + \bar{\mathcal{K}}_{pp}^2(t,s)$$

$$\overset{\text{def}}{=} \gamma_2^2 B \int_{\sigma=0}^{\infty} \bar{\Phi}_{11}^{\sigma}(t,s)\,\mathrm{d}\mu_{pp}(\sigma^2) + \gamma_1^2 B \int_{\sigma=0}^{\infty} \bar{\Phi}_{12}^{\sigma}(t,s)\,\mathrm{d}\mu_{pp}(\sigma^2)$$

$$= \gamma_2^2 B \int_{\sigma=0}^{1} \sigma^{3-\frac{1}{\alpha}} \bar{\Phi}_{11}^{\sigma}(t,s)\,\mathrm{d}\sigma + B \int_{\sigma=0}^{1} \sigma^{3-\frac{1}{\alpha}} \bar{\Phi}_{12}^{\sigma}(t,s)\,\mathrm{d}\sigma.$$

In the rest of this section we will provide upper-bounds on $\bar{\mathcal{K}}_{pp}$. We will divide the integral on $\sigma$ in different parts, as explained in Table 11. We will show in the following that $\bar{\mathcal{K}}_{pp}$ provides an upper-bound on the kernel function using our estimates on $\mu_{\mathcal{K}}$. We think this upper-bound is in fact tight but we cannot (and don't need to) entirely prove it with our current estimates on $\mu_{\mathcal{K}}$ from Proposition D.14 due to the oscillatory nature of $\Phi_{11}, \Phi_{12}$ in Proposition H.7. Instead we will rely on a slightly different argument for the lower-bound.

### H.10.1 First term: SGD noise

We will use the asymptotic of $\Phi_{11}^{\sigma}(t,s)$ developed above to show the following bounds on $\bar{\mathcal{K}}_{pp}^1(t,s)$:

**Proposition H.17.** *Suppose $\alpha > \frac{1}{4}$, $\alpha \neq \frac{1}{2}$, $\delta > \max\{4 - \frac{1}{\alpha}, 1\}$, $2\delta \notin \mathbb{N}$. Consider Parametrization H.1 with $\kappa_1 = (1-2\alpha)_+$, $0 \leq \kappa_b \leq \kappa_1$, $\kappa_2 = \kappa_1 - \kappa_b + 1$ and let any $M > 0$. There exists a constant $C(\alpha, H, M)$ such that for any $(s,t) \in \mathbb{R}_+^2$ with $s \leq t$, we have the bound :*

$$\bar{\mathcal{K}}_{pp}^1(t,s) \leq C \Bigg( \min\{\gamma_2^2 B \left(\sqrt{\gamma_3 B}(t-s)\right)^{-4+\frac{1}{\alpha}}, \gamma_2^2 B (\gamma_2 B(t-s))^{-2+\frac{1}{2\alpha}}\}$$

$$+ \gamma_2^2 B \left(\frac{1+t}{1+s}\right)^{-\delta} (\gamma_2 B(t-s))^{-2+\frac{1}{2\alpha}} \Bigg).$$

*Proof.* The proof is divided in three sub-cases depending on the relative size of $s, t, \frac{\gamma_2}{\gamma_3}$. Introduce some $\epsilon_1 > 0$ small enough.

**1st case:** $\frac{\gamma_2}{\gamma_3} \lesssim 1 + s \leq 1 + t$

We will suppose in the following $\sqrt{\gamma_3 B}(1+t) \gtrsim \sqrt{\gamma_3 B}(1+s) \gtrsim 1$ and $\gamma_2 B(1+t) \gtrsim \gamma_2 B(1+s) \gtrsim 1$. We additionally first assume that $\frac{\sqrt{\gamma_3 B}}{\gamma_2 B} \lesssim 1$. The other cases can be handled similarly by capping the bounds of the integrals. We hence decompose

$$\gamma_2^2 B \int_{\sigma=0}^{1} \sigma^{3-\frac{1}{\alpha}} \Phi_{11}(t,s)\,\mathrm{d}\sigma = \gamma_2^2 B \int_{\sigma=0}^{\frac{1}{\sqrt{\gamma_3 Bt}}} \sigma^{3-\frac{1}{\alpha}} \Phi_{11}(t,s)\,\mathrm{d}\sigma$$

$$+ \gamma_2^2 B \int_{\frac{1}{\sqrt{\gamma_3 Bt}}}^{\frac{1}{\sqrt{\gamma_3 Bs}}} \sigma^{3-\frac{1}{\alpha}} \Phi_{11}(t,s)\,\mathrm{d}\sigma + \gamma_2^2 B \int_{\frac{1}{\sqrt{\gamma_3 Bs}}}^{(1-\epsilon_1)\frac{\sqrt{4\gamma_3 B}}{\gamma_2 B}} \sigma^{3-\frac{1}{\alpha}} \Phi_{11}(t,s)\,\mathrm{d}\sigma$$

$$+ \gamma_2^2 B \int_{(1-\epsilon_1)\frac{\sqrt{4\gamma_3 B}}{\gamma_2 B}}^{(1+\epsilon_1)\frac{\sqrt{4\gamma_3 B}}{\gamma_2 B}} \sigma^{3-\frac{1}{\alpha}} \Phi_{11}(t,s)\,\mathrm{d}\sigma + \gamma_2^2 B \int_{(1+\epsilon_1)\frac{\sqrt{4\gamma_3 B}}{\gamma_2 B}}^{1} \sigma^{3-\frac{1}{\alpha}} \Phi_{11}(t,s)\,\mathrm{d}\sigma.$$

Continuing, we have

$$\gamma_2^2 B \int_{\sigma=0}^{1} \sigma^{3-\frac{1}{\alpha}} \Phi_{11}(t,s)\,\mathrm{d}\sigma \lesssim \gamma_2^2 B \int_{\sigma=0}^{\frac{1}{\sqrt{\gamma_3 Bt}}} \sigma^{3-\frac{1}{\alpha}} \times 1\,\mathrm{d}\sigma$$

$$+ \gamma_2^2 B \int_{\frac{1}{\sqrt{\gamma_3 Bt}}}^{\frac{1}{\sqrt{\gamma_3 Bs}}} \sigma^{3-\frac{1}{\alpha}} e^{-\gamma_2 B\sigma^2 t} \left( \sigma\sqrt{\gamma_3 B}(1+t) \right)^{-\delta}\,\mathrm{d}\sigma$$

$$+ \gamma_2^2 B \int_{\frac{1}{\sqrt{\gamma_3 Bs}}}^{(1-\epsilon_1)\frac{\sqrt{4\gamma_3 B}}{\gamma_2 B}} \sigma^{3-\frac{1}{\alpha}} e^{-\gamma_2 B\sigma^2(t-s)} \left( \frac{1+t}{1+s} \right)^{-\delta}\,\mathrm{d}\sigma$$

$$+ \gamma_2^2 B \int_{(1-\epsilon_1)\frac{\sqrt{4\gamma_3 B}}{\gamma_2 B}}^{(1+\epsilon_1)\frac{\sqrt{4\gamma_3 B}}{\gamma_2 B}} \sigma^{3-\frac{1}{\alpha}} \mathcal{O}(e^{-\frac{c_{\epsilon_1}(t-s)\gamma_3}{\gamma_2}})\,\mathrm{d}\sigma$$

$$+ \gamma_2^2 B \int_{(1+\epsilon_1)\frac{\sqrt{4\gamma_3 B}}{\gamma_2 B}}^{1} \sigma^{3-\frac{1}{\alpha}} \left( e^{-c_{\epsilon_1}\gamma_2 B\sigma^2(t-s)} + (x^4 + \frac{x^2}{(\gamma_2\sigma^2 s)^2} \right.$$

$$\left. + \frac{1}{(\gamma_2\sigma^2 t)^2}) e^{-c_{\epsilon_1}\frac{t-s}{d}} \left( \frac{1+t}{1+s} \right)^{\delta_2} \right)\,\mathrm{d}\sigma$$

$$\lesssim \gamma_2^2 B \left( \sqrt{\gamma_3 Bt} \right)^{-4+\frac{1}{\alpha}} \int_{u=0}^{1} u^{3-\frac{1}{\alpha}}\,\mathrm{d}u$$

$$+ \gamma_2^2 B \left( \sqrt{\gamma_3 Bt} \right)^{-4+\frac{1}{\alpha}} \int_{u=1}^{\frac{t}{s}} u^{3-\frac{1}{\alpha}-\delta} e^{-u^2\frac{\gamma_2}{\gamma_3 t}}\,\mathrm{d}u$$

$$+ \gamma_2^2 B \left( \frac{1+t}{1+s} \right)^{-\delta} (\gamma_2 B(t-s))^{-2+\frac{1}{2\alpha}} \int_{u=\frac{\gamma_2(t-s)}{\gamma_3 s^2}}^{\frac{\gamma_3}{\gamma_2}(t-s)(1-\epsilon_1)^2} u^{3-1/\alpha} e^{-u^2}\,\mathrm{d}u$$

$$+ \mathcal{O}(\left( \epsilon_1\frac{\sqrt{\gamma_3 B}}{\gamma_2 B} \right)^{4-\frac{1}{\alpha}} \gamma_2^2 B e^{-\frac{c_{\epsilon_1}(t-s)\gamma_3}{\gamma_2}})$$

$$+ \gamma_2^2 B(\gamma_2 B(t-s))^{-2+1/(2\alpha)} \left( \int_{u=\sqrt{\frac{\gamma_3(t-s)(1+\epsilon_1)}{\gamma_2}}}^{\sqrt{\gamma_2 B(t-s)}} u^{3-\frac{1}{\alpha}} e^{-u^2}\,\mathrm{d}u \right)$$

$$\lesssim \gamma_2^2 B \left( \sqrt{\gamma_3 Bt} \right)^{-4+\frac{1}{\alpha}}$$

$$+ \gamma_2^2 B \left( \frac{1+t}{1+s} \right)^{-\delta} (\gamma_2 B(t-s))^{-2+\frac{1}{2\alpha}} \begin{cases} \left( \frac{(t-s)\gamma_3}{\gamma_2} \right)^{4-1/\alpha} & \text{if } t-s < \frac{\gamma_2}{\gamma_3} \\ C & \text{if } \frac{\gamma_2}{\gamma_3} < t-s < \frac{s^2\gamma_3}{\gamma_2} \\ e^{-\frac{(t-s)\gamma_2}{\gamma_3 s^2}} & \text{if } \frac{s^2\gamma_3}{\gamma_2} < t-s \end{cases}$$

$$+ \gamma_2^2 B(\gamma_2 B(t-s))^{-2+1/(2\alpha)} \begin{cases} (\gamma_2 B(t-s))^{2-1/(2\alpha)} & \text{if } \gamma_2 B(t-s) < 1 \\ C & \text{if } \frac{1}{\gamma_2 B} < t-s < \frac{\gamma_2}{\gamma_3} \\ e^{-\frac{(t-s)\gamma_3}{\gamma_2}} & \text{if } \frac{\gamma_2}{\gamma_3} < t-s. \end{cases}$$

We have made the change of variable $u = \sigma\sqrt{\gamma_3 B}(1+t)$ in the first and second integrals. We have used $\alpha > \frac{1}{4}$ for the first integral to converge. In the last integral we used $u^2 = \gamma_2 B\sigma^2(t-s)$ and noticed that the integral on $u$ vanishes as $\frac{\gamma_2(t-s)}{\gamma_3 s^2} \to \infty$, ie as $s < \sqrt{t\frac{\gamma_2}{\gamma_3}}$. We check that this result is equivalent up to a constant to the result in Proposition H.17.

We additionally bounded at the singular point for $t - s \geq \frac{\gamma_2}{\gamma_3}$ and hence $\frac{1+t}{1+s} \lesssim \frac{(t-s)\gamma_3}{\gamma_2}$,

$$\gamma_2^2 B \int_{(1-\epsilon_1)\frac{\sqrt{4\gamma_3 B}}{\gamma_2 B}}^{(1+\epsilon_1)\frac{\sqrt{4\gamma_3 B}}{\gamma_2 B}} \sigma^{3-\frac{1}{\alpha}} \mathcal{O}(e^{-\frac{c_{\epsilon_1}(t-s)\gamma_3}{\gamma_2}})\,d\sigma$$

$$\lesssim \gamma_2^2 B \left(\frac{c_{\epsilon_1}(t-s)\gamma_3}{\gamma_2}\right)^{-\delta-2+1/(2\alpha)} \times \left(\frac{\sqrt{\gamma_3 B}}{\gamma_2 B}\right)^{4-1/\alpha}$$

$$\lesssim \left(\frac{1+t}{1+s}\right)^{-\delta}(\gamma_2 B(t-s))^{-2+1/(2\alpha)}.$$

For $t - s \leq \frac{\gamma_2}{\gamma_3}$ we have $\frac{1+t}{1+s} \lesssim 1$ because $1 + s \geq \frac{\gamma_2}{\gamma_3}$ which brings the upper-bound directly. In both cases, this term is absorbed by the other integrals.

We additionally observed that the following term is also absorbed by the others

$$\gamma_2^2 B \int_{(1+\epsilon_1)\frac{\sqrt{4\gamma_3 B}}{\gamma_2 B}}^{1} \sigma^{3-\frac{1}{\alpha}}\left(x^4 + \frac{x^2}{(\gamma_2 B\sigma^2 s)^2} + \frac{1}{(\gamma_2 B\sigma^2 t)^2}\right)e^{-c_{\epsilon_1}\frac{(t-s)\gamma_3}{\gamma_2}}\left(\frac{1+t}{1+s}\right)^{\delta_2}\,d\sigma.$$

Indeed, we know that $1 + t \geq 1 + s \geq \frac{\gamma_2}{\gamma_3} \gtrsim \frac{\gamma_3}{\gamma_2 B}$. This implies that

$$\gamma_2 B\sigma^2 t \gtrsim \gamma_2 B\sigma^2 s \gtrsim \left(\frac{\gamma_2 B\sigma}{\sqrt{\gamma_3 B}}\right)^{-2} = x^{-2},$$

hence we only need to bound

$$\gamma_2^2 B \int_{(1+\epsilon_1)\frac{\sqrt{4\gamma_3 B}}{\gamma_2 B}}^{1} \sigma^{3-\frac{1}{\alpha}}x^4 e^{-c_{\epsilon_1}\frac{(t-s)\gamma_3}{\gamma_2}}\left(\frac{1+t}{1+s}\right)^{\delta_2}\,d\sigma$$

$$\lesssim \gamma_2^2 B \left(\frac{\gamma_2 B}{\sqrt{\gamma_3 B}}\right)^{-4}\int_{(1+\epsilon_1)\frac{\sqrt{4\gamma_3 B}}{\gamma_2 B}}^{1} \sigma^{-1-\frac{1}{\alpha}}\,d\sigma \times e^{-c_{\epsilon_1}\frac{(t-s)\gamma_3}{\gamma_2}}\left(\frac{1+t}{1+s}\right)^{-2\delta}$$

$$\lesssim \gamma_2^2 B \left(\frac{\gamma_2 B}{\sqrt{\gamma_3 B}}\right)^{-4+1/\alpha} \times e^{-c_{\epsilon_1}\frac{(t-s)\gamma_3}{\gamma_2}}\left(\frac{1+t}{1+s}\right)^{-2\delta}$$

$$\lesssim \gamma_2^2 B(\gamma_2 B(t-s))^{-2+1/(2\alpha)}\left(\frac{1+t}{1+s}\right)^{-\delta}.$$

We used when $t - s \gtrsim \frac{\gamma_2}{\gamma_3}$ that for $\forall \eta > 0, \forall y \geq 1,\ e^{-y} \lesssim y^{-\eta}$ with $\eta = -2 + \frac{1}{2\alpha}$. For $t - s \lesssim \frac{\gamma_2}{\gamma_3}$ we still have $\gamma_2^2 B \left(\frac{\gamma_2 B}{\sqrt{\gamma_3 B}}\right)^{-4+1/\alpha} \lesssim \gamma_2^2 B(\gamma_2 B(t-s))^{-2+\frac{1}{2\alpha}}$.

**2nd case:** $1 + s \leq 1 + t \leq \frac{\gamma_2}{\gamma_3}$ We adopt the same strategy although we will go faster as some computations are very similar

$$\gamma_2^2 B \int_0^1 \sigma^{3-\frac{1}{\alpha}} \Phi_{11}(t,s) \, d\sigma \lesssim \gamma_2^2 B \int_0^{(1+\epsilon_1)\frac{\sqrt{4\gamma_3 B}}{\gamma_2 B}} \sigma^{3-\frac{1}{\alpha}} \times 1 \times d\sigma$$

$$+ \gamma_2^2 B \int_{(1+\epsilon_1)\frac{\sqrt{4\gamma_3 B}}{\gamma_2 B}}^{\frac{1}{\sqrt{\gamma_2 Bt}}} \sigma^{3-\frac{1}{\alpha}} \times 1 \times d\sigma$$

$$+ \gamma_2^2 B \int_{\frac{1}{\sqrt{\gamma_2 Bt}}}^{\frac{1}{\sqrt{\gamma_2 Bs}}} \sigma^{3-\frac{1}{\alpha}} e^{-\gamma_2 B\sigma^2 t} \, d\sigma + \gamma_2^2 B \int_{\frac{1}{\sqrt{\gamma_2 Bs}}}^{1} \sigma^{3-\frac{1}{\alpha}} \left( e^{-\gamma_2 B\sigma^2(t-s)} \right.$$

$$+ \left( x^4 + \frac{x^2}{(\gamma_2 B\sigma^2 s)^2} + \frac{1}{(\gamma_2 B\sigma^2 t)^2} \right) e^{-c_{\epsilon_1}\frac{(t-s)\gamma_3}{\gamma_2}} \left( \frac{1+t}{1+s} \right)^{\delta_2} \right) d\sigma$$

$$\lesssim \gamma_2^2 B (\gamma_2 Bt)^{-2+1/(2\alpha)} \int_{u=0}^{\frac{t}{s}} u^{3-\frac{1}{\alpha}} e^{-u^2} \, du$$

$$+ \gamma_2^2 B (\gamma_2 B(t-s))^{-2+1/(2\alpha)} \int_{u=\frac{t-s}{s}}^{\sqrt{\gamma_2 B(t-s)}} u^{3-\frac{1}{\alpha}} e^{-u^2} \, du$$

$$\lesssim \gamma_2^2 B (\gamma_2 Bt)^{-2+1/(2\alpha)}$$

$$+ \gamma_2^2 B (\gamma_2 B(t-s))^{-2+1/(2\alpha)} \begin{cases} (\gamma_2 B(t-s))^{2-1/(2\alpha)} & \text{if } \gamma_2 B(t-s) < 1 \\ C & \text{if } \frac{1}{\gamma_2 B} < t-s < s \\ e^{-\frac{t-s}{s}} & \text{if } s < t-s. \end{cases}$$

where in the first 3 integrals we used the change of variable $u = \sqrt{\gamma_2 B\sigma^2 t}$, that $e^{-u^2} \asymp 1$ for $\sigma < \frac{1}{\sqrt{\gamma_2 Bt}}$ and that $\alpha > \frac{1}{4}$ for convergence of the integral for small $u$. In the last integral, we made the change of variable $u = \sqrt{\gamma_2 B\sigma^2(t-s)}$.

We additionally noticed that

$$\int_{\frac{1}{\sqrt{\gamma_2 Bs}}}^{1} \sigma^{3-\frac{1}{\alpha}} \left( \frac{\sqrt{\gamma_3 B}}{\gamma_2 B\sigma} \right)^4 \left( \frac{1+t}{1+s} \right)^{\delta_2} d\sigma$$

$$\lesssim \left( \frac{\sqrt{\gamma_3 B}}{\gamma_2 B} \right)^4 \int_{\frac{1}{\sqrt{\gamma_2 Bs}}}^{2} \sigma^{-1-1/\alpha} \, d\sigma \left( \frac{1+t}{1+s} \right)^{\delta_2}$$

$$\lesssim \left( \frac{\gamma_3}{\gamma_2(\gamma_2 B)} \right)^2 (\gamma_2 Bs)^{\frac{1}{2\alpha}} \left( \frac{1+t}{1+s} \right)^{\delta_2}$$

$$\lesssim (\gamma_2 Bt)^{-2+\frac{1}{2\alpha}}$$

for $\delta$ large enough. **3rd case:** $1 + s \leq \frac{\gamma_2}{\gamma_3} \leq 1 + t$

$$\gamma_2^2 B \int_0^1 \sigma^{3-\frac{1}{\alpha}} \Phi_{11}(t,s)\,d\sigma \lesssim \gamma_2^2 B \int_0^{\frac{1}{\sqrt{\gamma_3 B t}}} \sigma^{3-\frac{1}{\alpha}} 1\,d\sigma$$

$$+ \gamma_2^2 B \int_{\frac{1}{\sqrt{\gamma_3 B t}}}^{(1-\epsilon_1)\frac{\sqrt{4\gamma_3 B}}{\gamma_2 B}} \sigma^{3-\frac{1}{\alpha}} e^{-\gamma_2 B \sigma^2 t} \left(\sigma\sqrt{\gamma_3 B}(1+t)\right)^{-\delta} d\sigma$$

$$+ \mathcal{O}^+\left(\gamma_2^2 B \int_{(1-\epsilon_1)\frac{\sqrt{4\gamma_3 B}}{\gamma_2 B}}^{(1+\epsilon_1)\frac{\sqrt{4\gamma_3 B}}{\gamma_2 B}} \sigma^{3-\frac{1}{\alpha}} e^{-\frac{\gamma_2 B \sigma^2}{2}(t-s)}\,d\sigma\right)$$

$$+ \gamma_2^2 B \int_{(1+\epsilon_1)\frac{\sqrt{4\gamma_3 B}}{\gamma_2 B}}^{\frac{1}{\sqrt{\gamma_2 B s}}} \sigma^{3-\frac{1}{\alpha}} e^{-\gamma_2 B \sigma^2 t}\,d\sigma + \gamma_2^2 B \int_{\frac{1}{\sqrt{\gamma_2 B s}}}^1 \sigma^{3-\frac{1}{\alpha}} e^{-\gamma_2 B \sigma^2 (t-s)}\,d\sigma$$

$$\lesssim \gamma_2^2 B \left(\sqrt{\gamma_3 B t}\right)^{-4+1/\alpha} \int_{u=0}^1 u^{3-1/\alpha}\,du$$

$$+ \gamma_2^2 B (\sqrt{\gamma_3 B}t)^{-4+1/\alpha} \int_{u=1}^{(1-\epsilon_1)\frac{\gamma_3 t}{\gamma_2}} u^{3-1/\alpha-\delta} e^{-u}\,du$$

$$+ \mathcal{O}^+\left(\left(\epsilon_1 \frac{\sqrt{\gamma_3 B}}{\gamma_2 B}\right)^{4-\frac{1}{\alpha}} e^{-\frac{(t-s)\gamma_3}{2\gamma_2}}\right)$$

$$+ \gamma_2^2 B (\gamma_2 B t)^{-2+1/(2\alpha)} \int_{u=(1+\epsilon_1)\sqrt{\frac{t\gamma_3}{\gamma_2}}}^{\sqrt{\frac{t}{s}}} u^{3-1/\alpha} e^{-u^2}\,du$$

$$+ \gamma_2^2 B (\gamma_2 B(t-s))^{-2+1/(2\alpha)} \int_{u=\sqrt{\frac{t-s}{s}}}^{\sqrt{\gamma_2 B(t-s)}} u^{3-1/\alpha} e^{-u^2}\,du$$

$$\lesssim \gamma_2^2 B \left(\sqrt{\gamma_3 B t}\right)^{-4+1/\alpha}$$

$$+ \gamma_2^2 B (\gamma_2 B(t-s))^{-2+1/(2\alpha)} \begin{cases} (\gamma_2 B(t-s))^{2-1/(2\alpha)} & \text{if} \quad \gamma_2 B(t-s) < 1 \\ C & \text{if} \quad \frac{1}{\gamma_2 B} < t-s < s \\ e^{-\frac{t-s}{s}} & \text{if} \quad s < t-s. \end{cases}$$

where we used in the first 2 integrals the change $u = \sigma\sqrt{\gamma_3 B}t$ and that $e^{-u} \asymp 1$ for $u \leq 1$.  $\qquad\square$

### H.10.2  Second term: momentum noise

We will use the asymptotics on $\Phi_{12}(t,s)$ to show the following:

**Proposition H.18.** *Suppose $\alpha > \frac{1}{4}$, $\alpha \neq \frac{1}{2}$, $\delta > \max\{4 - \frac{1}{\alpha}, 1\}$, $2\delta \notin \mathbb{N}$. Consider Parametrization H.1 with $\kappa_1 = (1-2\alpha)_+$, $0 \leq \kappa_b \leq \kappa_1$, $\kappa_2 = \kappa_1 - \kappa_b + 1$ and let any $M > 0$. There exists a constant $C(\alpha, H, M)$ such that for any $(s,t) \in \mathbb{R}_+^2$ with $s \leq t$, we have the bounds*

$$\begin{cases} \bar{\bar{\mathcal{K}}}_{pp}^2(t,s) \lesssim \bar{\bar{\mathcal{K}}}_{pp}^1(t,s) \text{ if } t-s \leq \frac{\gamma_2}{\gamma_3} \text{ or } s \leq \frac{\gamma_2}{\gamma_3} \\ \bar{\mathcal{K}}_{pp}^2(t,s) \leq C\left(\gamma_2^2 B(\sqrt{\gamma_3 B}(1+t))^{-4+\frac{1}{\alpha}} \left(\frac{(1+s)\gamma_3}{\gamma_2}\right)^2 + \gamma_3 \left(\frac{1+t}{1+s}\right)^{-\delta} (\gamma_2 B t)^{-1+1/(2\alpha)}\right) \\ \quad \text{if } 1+s \geq \frac{\gamma_2}{\gamma_3}, t-s \geq \frac{\gamma_2}{\gamma_3}. \end{cases}$$

*Proof.* The proof is again divided in 3 sub-cases. Let $\epsilon_1 > 0$ small enough, $\alpha > \frac{1}{4}$ and $\delta > 4 - \frac{1}{\alpha}$.

**1st case:** $\frac{\gamma_2}{\gamma_3} \leq 1+s \leq 1+t$

$$B \int_{\sigma=0}^{1} \sigma^{3-\frac{1}{\alpha}} \Phi_{12}(t,s) \, d\sigma = B \int_{\sigma=0}^{\frac{1}{\sqrt{\gamma_3 B t}}} \sigma^{3-\frac{1}{\alpha}} \Phi_{12}(t,s) \, d\sigma + B \int_{\frac{1}{\sqrt{\gamma_3 B t}}}^{\frac{1}{\sqrt{\gamma_3 B s}}} \sigma^{3-\frac{1}{\alpha}} \Phi_{12}(t,s) \, d\sigma$$

$$+ B \int_{\frac{1}{\sqrt{\gamma_3 B s}}}^{(1-\epsilon_1)\frac{\sqrt{4\gamma_3 B}}{\gamma_2 B}} \sigma^{3-\frac{1}{\alpha}} \Phi_{12}(t,s) \, d\sigma + B \int_{(1-\epsilon_1)\frac{\sqrt{4\gamma_3 B}}{\gamma_2 B}}^{(1+\epsilon_1)\frac{\sqrt{4\gamma_3 B}}{\gamma_2 B}} \sigma^{3-\frac{1}{\alpha}} \Phi_{12}(t,s) \, d\sigma$$

$$+ B \int_{(1+\epsilon_1)\frac{\sqrt{4\gamma_3 B}}{\gamma_2 B}}^{1} \sigma^{3-\frac{1}{\alpha}} \Phi_{12}(t,s) \, d\sigma$$

$$\lesssim B \int_{\sigma=0}^{\frac{1}{\sqrt{\gamma_3 B t}}} \sigma^{3-\frac{1}{\alpha}} \frac{\gamma_3}{B\sigma^2} (\sigma\sqrt{\gamma_3 B}(1+t))^2 \left(\frac{1+t}{1+s}\right)^{-2} d\sigma$$

$$+ B \int_{\frac{1}{\sqrt{\gamma_3 B t}}}^{\frac{1}{\sqrt{\gamma_3 B s}}} \sigma^{3-\frac{1}{\alpha}} \frac{\gamma_3}{B\sigma^2} (\sigma\sqrt{\gamma_3 B}(1+t))^{-\delta} e^{-\gamma_2 B\sigma^2 t} (\sigma\sqrt{\gamma_3 B}(1+s))^2 \, d\sigma$$

$$+ B \int_{\frac{1}{\sqrt{\gamma_3 B s}}}^{(1-\epsilon_1)\frac{\sqrt{4\gamma_3 B}}{\gamma_2 B}} \sigma^{3-\frac{1}{\alpha}} \frac{\gamma_3}{B\sigma^2} e^{-\gamma_2 B\sigma^2(t-s)} \left(\frac{1+t}{1+s}\right)^{-\delta} d\sigma$$

$$+ \int_{(1-\epsilon_1)\frac{\sqrt{4\gamma_3 B}}{\gamma_2 B}}^{(1+\epsilon_1)\frac{\sqrt{4\gamma_3 B}}{\gamma_2 B}} \sigma^{3-\frac{1}{\alpha}} \frac{\gamma_3}{B\sigma^2} \mathcal{O}^+ \left(e^{-\gamma_2 B\sigma^2(t-s)}\right) d\sigma$$

$$+ B \int_{(1+\epsilon_1)\frac{\sqrt{4\gamma_3 B}}{\gamma_2 B}}^{1} \sigma^{3-\frac{1}{\alpha}} \frac{\gamma_3}{B\sigma^2} \Bigg( \mathcal{O}(x^2 + (\gamma_2 B\sigma^2(1+s))^{-2}) e^{\mu_1(t-s)} \left(\frac{1+t}{1+s}\right)^{\delta_1}$$

$$+ \mathcal{O}(x^2 + (\gamma_2 B\sigma^2(1+t))^{-2}) e^{\mu_2(t-s)} \left(\frac{1+t}{1+s}\right)^{\delta_2}$$

$$+ \mathcal{O}(x^2 + x(\gamma_2 B\sigma^2(1+s))^{-1} + (\gamma_2 B\sigma^2(1+t))^{-2}) e^{\mu_3(t-s)} \left(\frac{1+t}{1+s}\right)^{\delta_3} \Bigg) d\sigma$$

$$\lesssim \gamma_3 \left(\frac{1+t}{1+s}\right)^{-2} (\sqrt{\gamma_3 B}(1+t))^{-2+\frac{1}{\alpha}} \int_{u=0}^{1} u^{3-\frac{1}{\alpha}} \, du$$

$$+ \gamma_3 \left(\frac{1+t}{1+s}\right)^{-2} (\sqrt{\gamma_3 B}(1+t))^{-2+\frac{1}{\alpha}} \int_{u=1}^{\frac{t}{s}} u^{3-\frac{1}{\alpha}-\delta} e^{-\frac{u^2\gamma_2}{\gamma_3 t}} \, du$$

$$+ \gamma_3 \left(\frac{1+t}{1+s}\right)^{-\delta} (\sqrt{\gamma_2 B(t-s)})^{-2+\frac{1}{\alpha}} \int_{u=\sqrt{\frac{\gamma_2(t-s)}{\gamma_3 s^2}}}^{\sqrt{\frac{t-s}{d}}} u^{1-\frac{1}{\alpha}} e^{-u^2} \, du$$

$$+ \gamma_3 (\gamma_2 B(t-s))^{-1+1/(2\alpha)} \int_{\sqrt{\frac{t-s}{d}}}^{\sqrt{\gamma_2 B(t-s)}} u^{-1-\frac{1}{\alpha}} e^{-2u^2} \, du$$

$$\lesssim \gamma_3 \left(\frac{1+t}{1+s}\right)^{-2} (\sqrt{\gamma_3 B}(1+t))^{-2+\frac{1}{\alpha}} + \gamma_3 \left(\frac{1+t}{1+s}\right)^{-\delta} (\sqrt{\gamma_2 B(t-s)})^{-2+\frac{1}{\alpha}}.$$

where we have made the change of variable $u = \sigma\sqrt{\gamma_3 B}(1+t)$ in the first and second integral, used that $\alpha > \frac{1}{4}$ for the first integral on $u$ to converge, that $\delta > 4 - \frac{1}{\alpha}$ for the second integral on $u$ to converge, and note that $\frac{\gamma_2}{\gamma_3 t} \ll 1$. Finally, in the last integral we used the change of variable $u \overset{\text{def}}{=} \sqrt{\gamma_2 B\sigma^2(t-s)}$.

Also note that we bounded the non-exponential term similarly as for $\mathcal{K}^1(t,s)$, ie since know that $1 + t \geq 1 + s \geq \frac{\gamma_2}{\gamma_3} \gtrsim \frac{\gamma_3}{\gamma_2 B}$, this implies that

$$\gamma_2 B\sigma^2 t \gtrsim \gamma_2 B\sigma^2 s \gtrsim \left(\frac{\gamma_2 B\sigma}{\sqrt{\gamma_3 B}}\right)^{-2} = x^{-2},$$

hence we only need to bound

$$B \int_{(1+\epsilon_1)\frac{\sqrt{4\gamma_3 B}}{\gamma_2 B}}^{1} \sigma^{3-\frac{1}{\alpha}} \frac{\gamma_3}{B\sigma^2} x^2 e^{-c_{\epsilon_1} \frac{(t-s)\gamma_3}{\gamma_2}} \left(\frac{1+t}{1+s}\right)^{\delta_2} d\sigma$$

$$\lesssim \gamma_3 \left(\frac{\gamma_2 B}{\sqrt{\gamma_3 B}}\right)^{-2} \int_{(1+\epsilon_1)\frac{\sqrt{4\gamma_3 B}}{\gamma_2 B}}^{1} \sigma^{-1-\frac{1}{\alpha}} d\sigma \times e^{-c_{\epsilon_1} \frac{(t-s)\gamma_3}{\gamma_2}} \left(\frac{1+t}{1+s}\right)^{-2\delta}$$

$$\lesssim \gamma_3 \left(\frac{\gamma_2 B}{\sqrt{\gamma_3 B}}\right)^{-2+1/\alpha} \times e^{-c_{\epsilon_1} \frac{(t-s)\gamma_3}{\gamma_2}} \left(\frac{1+t}{1+s}\right)^{-2\delta}$$

$$\lesssim \gamma_3 (\gamma_2 B(t-s))^{-2+1/(2\alpha)} \left(\frac{1+t}{1+s}\right)^{-\delta}.$$

**2nd case:** $1 + s \leq \frac{\gamma_2}{\gamma_3} \leq 1 + t$

In that case,

$$\bar{\mathcal{K}}_{pp}^2(t,s) = B \int_0^1 \sigma^{3-\frac{1}{\alpha}} \Phi_{12}(t,s) d\sigma \lesssim B \int_0^{\frac{1}{\sqrt{\gamma_3 B t}}} \sigma^{3-\frac{1}{\alpha}} \gamma_3^2 (1+s)^2 d\sigma$$

$$+ B \int_{\frac{1}{\sqrt{\gamma_3 B t}}}^{(1-\epsilon_1)\frac{\sqrt{4\gamma_3 B}}{\gamma_2 B}} \sigma^{3-\frac{1}{\alpha}} e^{-\gamma_2 B \sigma^2 t} (\sigma\sqrt{\gamma_3 B}(1+t))^{-\delta} \gamma_3^2 (1+s)^2 d\sigma$$

$$+ B \int_{(1-\epsilon_1)\frac{\sqrt{4\gamma_3 B}}{\gamma_2 B}}^{(1+\epsilon_1)\frac{\sqrt{4\gamma_3 B}}{\gamma_2 B}} \sigma^{3-\frac{1}{\alpha}} \mathcal{O}^+ \left(e^{-\frac{\gamma_2 B \sigma^2 (t-s)}{2}} d\sigma\right)$$

$$+ B\gamma_2^2 \int_{(1+\epsilon_1)\frac{\sqrt{4\gamma_3 B}}{\gamma_2 B}}^{\frac{1}{\sqrt{\gamma_2 B s}}} \sigma^{3-\frac{1}{\alpha}} (\sqrt{\gamma_3 B}\sigma(1+s))^2 e^{-2\gamma_2 B \sigma^2 t} d\sigma$$

$$+ \mathcal{O}^+ \left(B \int_{\frac{1}{\sqrt{\gamma_2 B s}}}^1 \sigma^{3-\frac{1}{\alpha}} \frac{\gamma_3}{B\sigma^2} \left(\frac{\sqrt{\gamma_3 B}}{\gamma_2 B \sigma}\right)^2 (e^{-2\gamma_2 B \sigma^2 (t-s)} + e^{-\frac{t-s}{s}}) d\sigma\right)$$

$$\lesssim B\gamma_2^2 \left(\frac{(1+s)\gamma_3}{\gamma_2}\right)^2 (\sqrt{\gamma_3 B}(1+t))^{-4+\frac{1}{\alpha}}$$

$$\lesssim \bar{\bar{\mathcal{K}}}^1(t,s).$$

**3rd case:** $s \leq t \leq \frac{\gamma_2}{\gamma_3}$ In that case we have:

$$\bar{\mathcal{K}}_{pp}^2(t,s) = B \int_0^1 \sigma^{3-\frac{1}{\alpha}} \Phi_{12}(t,s) d\sigma \lesssim B \int_0^{(1+\epsilon_1)\frac{\sqrt{4\gamma_3 B}}{\gamma_2 B}} \sigma^{3-\frac{1}{\alpha}} \gamma_3^2 (1+s)^2 d\sigma$$

$$+ B \int_{(1+\epsilon_1)\frac{\sqrt{4\gamma_3 B}}{\gamma_2 B}}^{\frac{1}{\sqrt{\gamma_2 B t}}} \sigma^{3-\frac{1}{\alpha}} \gamma_3^2 (1+s)^2 d\sigma$$

$$+ B \int_{\frac{1}{\sqrt{\gamma_2 B t}}}^{\frac{1}{\sqrt{\gamma_2 B s}}} \sigma^{3-\frac{1}{\alpha}} \frac{\gamma_3}{B\sigma^2} (\gamma_2 B \sigma^2 s)^2 e^{-2\gamma_2 B \sigma^2 t} d\sigma$$

$$+ \mathcal{O}^+ \left(B \int_{\frac{1}{\sqrt{\gamma_2 B s}}}^1 \sigma^{3-\frac{1}{\alpha}} \frac{\gamma_3}{B\sigma^2} \left(\frac{\sqrt{\gamma_3 B}}{\gamma_2 B \sigma}\right)^2 (e^{-2\gamma_2 B \sigma^2 (t-s)} + e^{-\frac{t-s}{s}}) d\sigma\right)$$

$$\lesssim B\gamma_2^2 \left(\frac{(1+s)\gamma_3}{\gamma_2}\right)^2 (\gamma_2 B t)^{-2+\frac{1}{2\alpha}} + \left(\frac{1+s}{1+t}\right)^2 \gamma_3 B \sqrt{\gamma_2 B t}^{-2+\frac{1}{\alpha}}$$

$$+ B\gamma_2^2 (\gamma_2 B(t-s))^{-2+\frac{1}{2\alpha}} \left(\frac{(t-s)\gamma_3}{\gamma_2}\right)^2 \left(\frac{s}{t-s}\right)^{\frac{1}{2\alpha}}$$

$$\lesssim \bar{\bar{\mathcal{K}}}^1(t,s).$$

$$\square$$

### H.10.3 Summary

What we have shown can be summarized as:

$$
\bar{\mathcal{K}}^1_{pp}(t,s) \lesssim \bar{\bar{\mathcal{K}}}^1_{pp}(t,s) \stackrel{\text{def}}{=} \begin{cases} \gamma_2^2 B \min\{1, (\gamma_2 B(t-s))^{-2+\frac{1}{2\alpha}}\} & \text{if} \quad s \leq \frac{\gamma_2}{\gamma_3}, t \leq 2\frac{\gamma_2}{\gamma_3} \\ \gamma_2^2 B(\sqrt{\gamma_3 B}(1+t))^{-4+\frac{1}{\alpha}} & \text{if} \quad s \leq \frac{\gamma_2}{\gamma_3}, 2\frac{\gamma_2}{\gamma_3} \leq t \\ \gamma_2^2 B \left(\frac{1+t}{1+s}\right)^{-\delta} \min\{1, (\gamma_2 B(t-s))^{-2+\frac{1}{2\alpha}}\} \\ \vee \gamma_2^2 B(\sqrt{\gamma_3 B}(1+t))^{-4+\frac{1}{\alpha}} \text{if} \quad \frac{\gamma_2}{\gamma_3} \leq s \leq t. \end{cases}
$$

$$
\bar{\mathcal{K}}^2_{pp}(t,s) \lesssim \begin{cases} \lesssim \bar{\bar{\mathcal{K}}}^1_{pp}(t,s) & \text{if} \quad s \leq \frac{\gamma_2}{\gamma_3} \quad \text{or} \quad t-s \leq \frac{\gamma_2}{\gamma_3} \\ \gamma_3 \left(\frac{1+t}{1+s}\right)^{-\delta} (\gamma_2 B(t-s))^{-1+\frac{1}{2\alpha}} \vee \gamma_2^2 B(\sqrt{\gamma_3 B}(1+t))^{-4+\frac{1}{\alpha}} \left(\frac{(1+s)\gamma_3}{\gamma_2}\right)^2 \\ \text{if} \quad s \geq \frac{\gamma_2}{\gamma_3}, t-s \geq \frac{\gamma_2}{\gamma_3}. \end{cases}
$$

Hence Proposition H.17, Proposition H.18 together lead to the following proposition:

**Proposition H.19.** *Let $\alpha \neq \frac{1}{2}$ with $\alpha > \frac{1}{4}$ and $\delta > \max\{1, 4 - \frac{1}{\alpha}\}$. Denote $\bar{\bar{\mathcal{K}}}_{pp}(t,s)$ the kernel defined for any $t \geq s \geq 0$,*

$$
\bar{\bar{\mathcal{K}}}_{pp} \stackrel{\text{def}}{=} \begin{cases} \gamma_2^2 B \min\{1, (\gamma_2 B(t-s))^{-2+\frac{1}{2\alpha}}\} & \text{if} \quad s \leq \frac{\gamma_2}{\gamma_3}, t \leq 2\frac{\gamma_2}{\gamma_3} \quad \text{or} \quad t-s \leq \frac{\gamma_2}{\gamma_3}, \\ \gamma_2^2 B(\sqrt{\gamma_3 B}(1+t))^{-4+\frac{1}{\alpha}} & \text{if} \quad s \leq \frac{\gamma_2}{\gamma_3}, 2\frac{\gamma_2}{\gamma_3} < t, \\ \gamma_3 \left(\frac{1+t}{1+s}\right)^{-\delta} (\gamma_2 B(t-s))^{-1+\frac{1}{2\alpha}} \vee \gamma_2^2 B(\sqrt{\gamma_3 B}(1+t))^{-4+\frac{1}{\alpha}} \left(\frac{1+s}{\frac{\gamma_2}{\gamma_3}}\right)^2 \\ \text{if} \quad s > \frac{\gamma_2}{\gamma_3}, t-s > \frac{\gamma_2}{\gamma_3}. \end{cases}
$$

*Then we know that $\forall M > 0, \exists C(\alpha, \beta, M, \delta), \forall 0 \leq s \leq t$ with $\max\{\gamma_2 B(1+t), (\sqrt{\gamma_3 B}(1+t))^2\} \leq Md^{2\alpha}$,*

$$
\mathcal{K}(t,s) \leq \bar{\mathcal{K}}(t,s) \stackrel{\text{def}}{=} \bar{\mathcal{K}}^1(t,s) + \bar{\mathcal{K}}^2(t,s) \leq C\bar{\bar{\mathcal{K}}}_{pp}(t,s).
$$

*Proof.* By non-negativity of $\mu_{\mathcal{K}}$, it is clear that $\forall t, s \geq 0$, $\mathcal{K}(t,s) \leq \bar{\mathcal{K}}(t,s)$. Additionally, the estimate we previously derived show that $\bar{\mathcal{K}}_{pp}(t,s) \lesssim \bar{\bar{\mathcal{K}}}_{pp}(t,s)$. Hence we only need to show that $\bar{\mathcal{K}}(t,s) \lesssim \bar{\mathcal{K}}_{pp}(t,s)$.

We constructed $\bar{\Phi}^\sigma_{11}(t,s), \bar{\Phi}^\sigma_{12}(t,s)$ exactly to ensure that we can apply Proposition D.14 by bounding their oscillatory part. Hence, Proposition D.14 brings the existence of $\tilde{M}, M_2 > 0$ and $C > 0$ such that

$$
\int_{\tilde{M}d^{-2\alpha}}^{\frac{1}{M}} (\gamma_2^2 B \bar{\Phi}^{\sqrt{u}}_{11}(t,s) + \gamma_1^2 B \bar{\Phi}^{\sqrt{u}}_{12}(t,s))\mu_{\mathcal{K}}(\mathrm{d}u)
$$

$$
\leq C \int_{M_2 d^{-2\alpha}}^{\frac{1}{M_2}} (\gamma_2^2 B \bar{\Phi}^{\sqrt{u}}_{11}(t,s) + \gamma_1^2 B \bar{\Phi}^{\sqrt{u}}_{12}(t,s))\mu_{\mathcal{K}_{pp}}(\mathrm{d}u).
$$

To conclude, we only need to show that the small and large eigenvalues $\sigma$ do not contribute too much to the kernel, i.e.

$$
\max \begin{cases} \int_0^{\tilde{M}d^{-2\alpha}} (\gamma_2^2 B \bar{\Phi}^{\sqrt{u}}_{11}(t,s) + \gamma_1^2 B \bar{\Phi}^{\sqrt{u}}_{12}(t,s))\mu_{\mathcal{K}}(\mathrm{d}u) \\ \int_{\frac{1}{M}}^\infty (\gamma_2^2 B \bar{\Phi}^{\sqrt{u}}_{11}(t,s) + \gamma_1^2 B \bar{\Phi}^{\sqrt{u}}_{12}(t,s))\mu_{\mathcal{K}}(\mathrm{d}u) \end{cases} \lesssim \bar{\mathcal{K}}_{pp}(t,s).
$$

The above follows immediately from Propositions D.8 to D.11 for $d$ large enough to apply Proposition D.9.

$$\square$$

## H.11 Verifying the hypothesis of Kesten's Lemma

The goal of this section is to show that the upper-bound on the kernel $\bar{\bar{\mathcal{K}}}$ in Proposition H.19 satisfies the hypothesis of Kesten's Lemma.

**Proposition H.20.** *Let $\alpha > \frac{1}{4}$, $\alpha \neq \frac{1}{2}$, $\delta > \max\{1, 4 - \frac{1}{\alpha}\}$, $2\delta \notin \mathbb{N}$. For any $M > 0$, $\epsilon > 0$, there exists some $C(\alpha, \delta, M, \epsilon)$ independent of $d$ such that considering Parametrization H.1 with $\kappa_1 \geq (1 - 2\alpha)_+$, $\kappa_2 \geq \kappa_1 - \kappa_b + 1$, $\kappa_b \leq \min\{\kappa_1, \kappa_2\}$, if $\max\{\tilde{\gamma}_2, \frac{\tilde{\gamma}_3}{\tilde{\gamma}_2}\} \leq C$ then $\forall 0 \leq s \leq t$ with $\max\{\gamma_2 B(1 + t), (\sqrt{\gamma_3 B}(1 + t))^2\} \leq M d^{2\alpha}$ we have*

$$\int_{r=s}^{t} \bar{\bar{\mathcal{K}}}_{pp}(t, r)\bar{\bar{\mathcal{K}}}_{pp}(r, s)\, \mathrm{d}r \leq \epsilon \bar{\bar{\mathcal{K}}}_{pp}(t, s).$$

*Proof.* This is a consequence of the form of $\bar{\bar{\mathcal{K}}}_{pp}$ computed above. We differentiate different cases.

**1st case:** $s \leq t \lesssim \frac{\gamma_2}{\gamma_3}$ **or** $t - s \lesssim \frac{\gamma_2}{\gamma_3}$ Then the proof sketch can be found in [80, Prop.G.2]. More precisely, we have

$$\int_{r=s}^{t} \bar{\bar{\mathcal{K}}}_{pp}(t, r)\bar{\bar{\mathcal{K}}}_{pp}(r, s)\, \mathrm{d}r \lesssim \int_{r=s}^{t} \gamma_2^2 B \overline{\gamma_2 B(t - r)}^{-2 + \frac{1}{2\alpha}} \gamma_2^2 B \overline{\gamma_2 B(r - s)}^{-2 + \frac{1}{2\alpha}}\, \mathrm{d}r$$

$$\lesssim \gamma_2^2 B \overline{\gamma_2 B(t - s)}^{-2 + \frac{1}{2\alpha}} \int_{r=s}^{t} \bar{\bar{\mathcal{K}}}_{pp}(t, r)\, \mathrm{d}r$$

$$\leq \epsilon \bar{\bar{\mathcal{K}}}_{pp}(t, s).$$

Here we used Lemma H.5 in the bound on the kernel integral.

**2nd case:** $s \leq \frac{\gamma_2}{\gamma_3}$ **and** $3\frac{\gamma_2}{\gamma_3} \leq t$ Then we write:

$$\int_{r=s}^{t} \bar{\bar{\mathcal{K}}}_{pp}(t, r)\bar{\bar{\mathcal{K}}}_{pp}(r, s)\, \mathrm{d}r \lesssim \int_{r=s}^{2\frac{\gamma_2}{\gamma_3}} \gamma_2^2 B \overline{(\gamma_2 B(r - s))^{-2 + \frac{1}{2\alpha}}} \times \gamma_2^2 B(\sqrt{\gamma_3 B}(1 + t))^{-4 + \frac{1}{\alpha}})\, \mathrm{d}r$$

$$+ \int_{r=2\frac{\gamma_2}{\gamma_3}}^{t - \frac{\gamma_2}{\gamma_3}} \left(\gamma_3 \left(\frac{1 + t}{1 + r}\right)^{-\delta} (\gamma_2 B(t - r))^{-1 + \frac{1}{2\alpha}} + \gamma_2^2 B(\sqrt{\gamma_3 B}(1 + t))^{-4 + \frac{1}{\alpha}} \left(\frac{(1 + r)\gamma_3}{\gamma_2}\right)^2\right)$$

$$\times \left(\gamma_2^2 B(\sqrt{\gamma_3 B}(1 + r))^{-4 + \frac{1}{\alpha}}\right)\, \mathrm{d}r$$

$$+ \int_{r=t - \frac{\gamma_2}{\gamma_3}}^{t} \gamma_2^2 B \sigma((\gamma_2 B(t - r))^{-2 + \frac{1}{2\alpha}}) \times \gamma_2^2 B(\sqrt{\gamma_3 B}(1 + r))^{-4 + \frac{1}{\alpha}})\, \mathrm{d}r$$

$$\lesssim \epsilon \gamma_2^2 B(\sqrt{\gamma_3 B}(1 + t))^{-4 + \frac{1}{\alpha}}.$$

Indeed it is clear for the first and last term. For the middle term we distribute and bound for $\tilde{\gamma}_2, \tilde{\gamma}_3$ small enough for a fixed $\epsilon$,

$$\int_{r=2\frac{\gamma_2}{\gamma_3}}^{t - \frac{\gamma_2}{\gamma_3}} \gamma_2^2 B(\sqrt{\gamma_3 B}(1 + t))^{-4 + \frac{1}{\alpha}} \left(\frac{(1 + r)\gamma_3}{\gamma_2}\right)^2 \times \gamma_2^2 B(\sqrt{\gamma_3 B}(1 + r))^{-4 + \frac{1}{\alpha}}\, \mathrm{d}r$$

$$\lesssim \gamma_2^2 B(\sqrt{\gamma_3 B}(1 + t))^{-4 + \frac{1}{\alpha}} \times \int_{r=2\frac{\gamma_2}{\gamma_3}}^{t - \frac{\gamma_2}{\gamma_3}} \left(\frac{(1 + r)\gamma_3}{\gamma_2}\right)^2 \gamma_2^2 B(\sqrt{\gamma_3 B}(1 + r))^{-4 + \frac{1}{\alpha}}\, \mathrm{d}r$$

$$\lesssim \epsilon \gamma_2^2 B(\sqrt{\gamma_3 B}(1 + t))^{-4 + \frac{1}{\alpha}}$$

because

$$\int_{r=2\frac{\gamma_2}{\gamma_3}}^{t-\frac{\gamma_2}{\gamma_3}} \left(\frac{(1+r)\gamma_3}{\gamma_2}\right)^2 \gamma_2^2 B(\sqrt{\gamma_3 B}(1+r))^{-4+\frac{1}{\alpha}} \,\mathrm{d}r$$

$$\lesssim \int_{r=2\frac{\gamma_2}{\gamma_3}}^{t-\frac{\gamma_2}{\gamma_3}} \frac{\gamma_3}{\gamma_2 B} \gamma_2 B(\sqrt{\gamma_3 B}(1+r))^{-2+\frac{1}{\alpha}} \,\mathrm{d}r$$

$$\lesssim d^{-1}\gamma_2 B d^{\alpha(-1+1/\alpha)_+}$$

$$\lesssim \epsilon.$$

And the other term,

$$\int_{r=2\frac{\gamma_2}{\gamma_3}}^{t-\frac{\gamma_2}{\gamma_3}} \gamma_3 \left(\frac{1+t}{1+r}\right)^{-\delta} (\gamma_2 B(t-r))^{-1+\frac{1}{2\alpha}} \times \gamma_2^2 B(\sqrt{\gamma_3 B}(1+r))^{-4+\frac{1}{\alpha}} \,\mathrm{d}r$$

$$\lesssim \int_{r=2\frac{\gamma_2}{\gamma_3}}^{t-\frac{\gamma_2}{\gamma_3}} \gamma_3 \left(\frac{1+t}{1+r}\right)^{-4+\frac{1}{\alpha}} (\gamma_2 B(t-r))^{-1+\frac{1}{2\alpha}} \times \gamma_2^2 B(\sqrt{\gamma_3 B}(1+r))^{-4+\frac{1}{\alpha}} \,\mathrm{d}r$$

$$\lesssim \gamma_2^2 B(\sqrt{\gamma_3 B}(1+t))^{-4+\frac{1}{\alpha}} \times \int_{\frac{\gamma_2}{\gamma_3}}^{t-\frac{\gamma_2}{\gamma_3}} \gamma_3(\gamma_2 B(t-r))^{-1+1/(2\alpha)} \,\mathrm{d}r$$

$$\lesssim \epsilon\gamma_2^2 B(\sqrt{\gamma_3 B}(1+t))^{-4+\frac{1}{\alpha}}$$

where we used the stability condition $\frac{\gamma_3}{\gamma_2 B} \lesssim \frac{1}{d}$.

**3rd case:** $s \geq \frac{\gamma_2}{\gamma_3}$ **and** $t-s \geq \frac{\gamma_2}{\gamma_3}$

$$\int_{r=s}^{t} \bar{\bar{\mathcal{K}}}_{pp}(t,r)\bar{\bar{\mathcal{K}}}_{pp}(r,s) \,\mathrm{d}r$$

$$\lesssim \int_{r=s}^{s+\frac{\gamma_2}{\gamma_3}} \left(\gamma_3 \left(\frac{1+t}{1+r}\right)^{-\delta} (\gamma_2 B(t-r))^{-1+\frac{1}{2\alpha}} + \gamma_2^2 B(\sqrt{\gamma_3 B}(1+t))^{-4+\frac{1}{\alpha}} \left(\frac{(1+r)\gamma_3}{\gamma_2}\right)^2\right)$$

$$\times \gamma_2^2 B\overline{\gamma_2 B(r-s)}^{-2+\frac{1}{2\alpha}} \,\mathrm{d}r$$

$$+ \int_{r=s+\frac{\gamma_2}{\gamma_3}}^{t-\frac{\gamma_2}{\gamma_3}} \left(\gamma_3 \left(\frac{1+t}{1+r}\right)^{-\delta} (\gamma_2 B(t-r))^{-1+\frac{1}{2\alpha}} + \gamma_2^2 B(\sqrt{\gamma_3 B}(1+t))^{-4+\frac{1}{\alpha}} \left(\frac{(1+r)\gamma_3}{\gamma_2}\right)^2\right)$$

$$\times \left(\gamma_3 \left(\frac{1+r}{1+s}\right)^{-\delta} (\gamma_2 B(r-s))^{-1+\frac{1}{2\alpha}} + \gamma_2^2 B(\sqrt{\gamma_3 B}(1+r))^{-4+\frac{1}{\alpha}} \left(\frac{(1+s)\gamma_3}{\gamma_2}\right)^2\right) \,\mathrm{d}r$$

$$+ \int_{r=t-\frac{\gamma_2}{\gamma_3}}^{t} \gamma_2^2 B\overline{\gamma_2 B(t-r)}^{-2+\frac{1}{2\alpha}}$$

$$\times \left(\gamma_3 \left(\frac{1+r}{1+s}\right)^{-\delta} (\gamma_2 B(r-s))^{-1+\frac{1}{2\alpha}} + \gamma_2^2 B(\sqrt{\gamma_3 B}(1+r))^{-4+\frac{1}{\alpha}} \left(\frac{(1+s)\gamma_3}{\gamma_2}\right)^2\right) \,\mathrm{d}r$$

$$\lesssim \epsilon \left(\gamma_3 \left(\frac{1+t}{1+s}\right)^{-\delta} (\gamma_2 B(t-s))^{-1+\frac{1}{2\alpha}} + \gamma_2^2 B(\sqrt{\gamma_3 B}(1+t))^{-4+\frac{1}{\alpha}} \left(\frac{(1+s)\gamma_3}{\gamma_2}\right)^2\right).$$

The last term is straightforward because of the stability condition on $\gamma_2$ (which implies that we can decrease $\tilde{\gamma}_2$ so that $\int_{r=t-\frac{\gamma_2}{\gamma_3}}^{t} \gamma_2^2 B\overline{\gamma_2 B(t-r)}^{-2+\frac{1}{2\alpha}} \,\mathrm{d}r \lesssim \int_{r=0}^{d^{2\alpha}/(\gamma_2 B)} \gamma_2^2 B\overline{\gamma_2 B(d^{2\alpha}-r)}^{-2+\frac{1}{2\alpha}} \,\mathrm{d}r \lesssim \epsilon$ and the fact that $t \asymp r$ in that integral. The first term is also straightforward for the same reason, ie stability condition on $\gamma_2$ which and the fact that $r \asymp s$ in the first integral. For the first term we develop and write

$$\int_{r=s}^{t-\frac{\gamma_2}{\gamma_3}} \gamma_3 \left(\frac{1+t}{1+r}\right)^{-\delta} (\gamma_2 B(t-r))^{-1+\frac{1}{2\alpha}} \times \gamma_3 \left(\frac{1+r}{1+s}\right)^{-\delta} (\gamma_2(r-s))^{-1+\frac{1}{2\alpha}} \, dr$$

$$\lesssim \gamma_3 \left(\frac{1+t}{1+s}\right)^{-\delta} (\gamma_2 B(t-s))^{-1+1/(2\alpha)} \int_{r=s}^{t-\frac{\gamma_2}{\gamma_3}} \gamma_3 (\gamma_2 B(r-s))^{-1+1/(2\alpha)} \, dr$$

$$\lesssim \epsilon \gamma_3 \left(\frac{1+t}{1+s}\right)^{-\delta} (\gamma_2 B(t-s))^{-1+\frac{1}{2\alpha}}.$$

Here we used the stability condition on $\gamma_3$ which implies that $\int_{r=s}^{t-\frac{\gamma_2}{\gamma_3}} \gamma_3 (\gamma_2 B(r-s))^{-1+1/(2\alpha)} \, dr \lesssim \int_{r=s}^{d^{2\alpha}/(\gamma_2 B)} \gamma_3 \overline{\gamma_2 B r}^{-1+1/(2\alpha)} \, dr \lesssim \epsilon$.

We additionnally bound

$$\int_{r=s+\frac{\gamma_2}{\gamma_3}}^{t-\frac{\gamma_2}{\gamma_3}} \left(\gamma_2^2 B(\sqrt{\gamma_3 B}(1+t))^{-4+\frac{1}{\alpha}} \left(\frac{(1+r)\gamma_3}{\gamma_2}\right)^2\right)$$

$$\times \left(\gamma_3 \left(\frac{1+r}{1+s}\right)^{-\delta} (\gamma_2 B(r-s))^{-1+\frac{1}{2\alpha}}\right) dr$$

$$\lesssim \int_{r=s+\frac{\gamma_2}{\gamma_3}}^{t-\frac{\gamma_2}{\gamma_3}} \left(\gamma_2^2 B(\sqrt{\gamma_3 B}(1+t))^{-4+\frac{1}{\alpha}} \left(\frac{(1+r)\gamma_3}{\gamma_2}\right)^2\right)$$

$$\times \left(\gamma_3 \left(\frac{1+r}{1+s}\right)^{-\delta} (\gamma_2 B(r-s))^{-1+\frac{1}{2\alpha}}\right) dr$$

$$\lesssim \gamma_2^2 B(\sqrt{\gamma_3 B}(1+t))^{-4+\frac{1}{\alpha}} \left(\frac{(1+s)\gamma_3}{\gamma_2}\right)^2$$

$$\int_{r=s+\frac{\gamma_2}{\gamma_3}}^{t-\frac{\gamma_2}{\gamma_3}} \left(\gamma_3 \left(\frac{1+r}{1+s}\right)^{-\delta+2} (\gamma_2 B(r-s))^{-1+\frac{1}{2\alpha}}\right) dr$$

$$\lesssim \epsilon \gamma_2^2 B(\sqrt{\gamma_3 B}(1+t))^{-4+\frac{1}{\alpha}} \left(\frac{(1+s)\gamma_3}{\gamma_2}\right)^2.$$

Here we used that $\delta > 2$ and the stability condition on $\gamma_3$ to get that $\gamma_3 \int_{r=s+\frac{\gamma_2}{\gamma_3}}^{t-\frac{\gamma_2}{\gamma_3}} (\gamma_2 B(r-s))^{-1+\frac{1}{2\alpha}} \, dr \lesssim \gamma_3 \int_{r=0}^{d^{2\alpha}/(\gamma_2 B)} \overline{\gamma_2 B r}^{-1+\frac{1}{2\alpha}} \, dr \lesssim \epsilon$.

We also write

$$\int_{r=s+\frac{\gamma_2}{\gamma_3}}^{t-\frac{\gamma_2}{\gamma_3}} \left(\gamma_2^2 B(\sqrt{\gamma_3 B}(1+t))^{-4+\frac{1}{\alpha}} \left(\frac{(1+r)\gamma_3}{\gamma_2}\right)^2\right)$$

$$\times \left(\gamma_2^2 B(\sqrt{\gamma_3 B}(1+r))^{-4+\frac{1}{\alpha}} \left(\frac{(1+s)\gamma_3}{\gamma_2}\right)^2\right) dr$$

$$\lesssim \gamma_2^2 B(\sqrt{\gamma_3 B}(1+t))^{-4+\frac{1}{\alpha}} \left(\frac{(1+s)\gamma_3}{\gamma_2}\right)^2 \int_{r=s+\frac{\gamma_2}{\gamma_3}}^{t-\frac{\gamma_2}{\gamma_3}} \frac{\gamma_3}{\gamma_2 B} \gamma_2 B(\sqrt{\gamma_3 B}(1+r))^{-2+\frac{1}{\alpha}} \, dr$$

$$\lesssim \epsilon \gamma_2^2 B(\sqrt{\gamma_3 B}(1+t))^{-4+\frac{1}{\alpha}} \left(\frac{(1+s)\gamma_3}{\gamma_2}\right)^2 d^{-1} \times \begin{cases} \gamma_2 B \times d^{\alpha(-1+\frac{1}{\alpha})} & \text{if } \alpha < 1 \\ \gamma_2 B \left(\frac{\gamma_2 B}{\sqrt{\gamma_3 B}}\right)^{-2+\frac{1}{\alpha}} & \text{if } \alpha > 1 \end{cases}$$

$$\lesssim \epsilon \gamma_2^2 B(\sqrt{\gamma_3 B}(1+t))^{-4+\frac{1}{\alpha}} \left(\frac{(1+s)\gamma_3}{\gamma_2}\right)^2.$$

Here we used the stability condition $\frac{\gamma_3}{\gamma_2 B} \lesssim d^{-1}$ and that $d^{-\alpha} \lesssim \frac{\sqrt{\gamma_3 B}}{\gamma_2 B} \lesssim 1$.

Finally we only have to bound the following term.

$$\int_{r=s}^{t-\frac{\gamma_2}{\gamma_3}} \gamma_3 \left(\frac{1+t}{1+r}\right)^{-\delta} (\gamma_2 B(t-r))^{-1+\frac{1}{2\alpha}} \times \gamma_2^2 B(\sqrt{\gamma_3 B}(1+r))^{-4+\frac{1}{\alpha}} \left(\frac{(1+s)\gamma_3}{\gamma_2}\right)^2 \mathrm{d}r$$

$$\lesssim \int_{r=s}^{t-\frac{\gamma_2}{\gamma_3}} \gamma_3 \left(\frac{1+t}{1+r}\right)^{-4+\frac{1}{\alpha}} (\gamma_2 B(t-r))^{-1+\frac{1}{2\alpha}}$$

$$\times \gamma_2^2 B(\sqrt{\gamma_3 B}(1+r))^{-4+\frac{1}{\alpha}} \left(\frac{(1+s)\gamma_3}{\gamma_2}\right)^2 \mathrm{d}r$$

$$\lesssim \gamma_2^2 B(\sqrt{\gamma_3 B}(1+t))^{-4+\frac{1}{\alpha}} \left(\frac{(1+s)\gamma_3}{\gamma_2}\right)^2 \times \int_{r=s}^{t-\frac{\gamma_2}{\gamma_3}} \gamma_3 (\gamma_2 B(t-r))^{-1+\frac{1}{2\alpha}} \mathrm{d}r$$

$$\lesssim \epsilon \gamma_2^2 B(\sqrt{\gamma_3 B}(1+t))^{-4+\frac{1}{\alpha}} \left(\frac{(1+s)\gamma_3}{\gamma_2}\right)^2.$$

where we used that $\delta > 4 - \frac{1}{\alpha}$, and the stability condition on $\gamma_3$ for the second integral to be small.

$\square$

Below we show an intermediary result that gives a lower-bound on the term $\mathcal{F} * \mathcal{K}$ of the Volterra equation solution.

**Proposition H.21.** *Let* $\alpha, \beta$ *with* $\alpha > \frac{1}{4}$ $\alpha, \beta \neq \frac{1}{2}$, $2\alpha + 2\beta > 1$, $\alpha + 1 > \beta$. *Then for any* $M > 0$, *there exists some* $C > 0$ *such that for any* $t \geq 0$, *if* $\max\{\gamma_2 Bt, (\sqrt{\gamma_3 B}t)^2\} \leq Md^{2\alpha}$ *and* $\gamma_2 Bt \geq 1$,

$$[\mathcal{F} * \mathcal{K}](t) \geq \frac{1}{C \times \gamma_2 B} \bar{\bar{\mathcal{K}}}_{pp}(t, 0).$$

*Proof.* We remind that under Parametrization H.1 $\frac{\sqrt{\gamma_3 B}}{\gamma_2 B} \lesssim 1$. This ensures that $\frac{1}{\gamma_2 B} \lesssim \frac{\gamma_2}{\gamma_3}$.

Using Propositions H.13, H.14 and H.16 we have precise asymptotics on the forcing function $\mathcal{F}(t)$ for all times. Using these it is clear that for any $T \geq 0$ with $\gamma_2 BT \geq 1$ we have $\int_0^T \mathcal{F}(s)\,\mathrm{d}s \gtrsim \frac{1}{\gamma_2 B}$.

We now differentiate the two cases $t \lesssim \frac{\gamma_2}{\gamma_3}$ and $t \gtrsim \frac{\gamma_2}{\gamma_3}$ for which $\bar{\bar{\mathcal{K}}}_{pp}(t, 0)$ shows two distinct behaviors.

For $t \gtrsim \frac{\gamma_2}{\gamma_3}$ we know that for any $s \geq 0$ with $s \leq t$,

$$\mathcal{K}(t, s) \gtrsim \int_{M^{-1/2}d^{-\alpha}}^{\frac{1}{\sqrt{\gamma_3 B}t}} \gamma_1^2 B \Phi_{12}^\sigma(t, s)\,\mathrm{d}\mu_{\mathcal{K}}(\mathrm{d}\sigma^2)$$

$$\gtrsim \int_{M^{-1/2}d^{-\alpha}}^{\frac{1}{\sqrt{\gamma_3 B}t}} \gamma_1^2 B \times \frac{\gamma_3}{B\sigma^2} \times \mathrm{d}\mu_{\mathcal{K}_{pp}}(\mathrm{d}\sigma^2)$$

$$\gtrsim \gamma_3 \overline{(\sqrt{\gamma_3 B}t)}^{-4+\frac{1}{\alpha}}$$

$$\gtrsim \bar{\bar{\mathcal{K}}}_{pp}(t, 0).$$

Similarly if $t \lesssim \frac{\gamma_2}{\gamma_3}$ we write for any $s \geq 0$ with $s \leq t$

$$\mathcal{K}(t, s) \gtrsim \int_{\max\{M^{-1/2}d^{-\alpha}, \frac{\sqrt{\gamma_3 B}}{\gamma_2 B}\}}^{\frac{1}{\sqrt{\gamma_2 B}t}} \gamma_2^2 B \Phi_{11}^\sigma(t, s)\,\mathrm{d}\mu_{\mathcal{K}}(\mathrm{d}\sigma^2)$$

$$\gtrsim \int_{\max\{M^{-1/2}d^{-\alpha}, \frac{\sqrt{\gamma_3 B}}{\gamma_2 B}\}}^{\frac{1}{\sqrt{\gamma_2 B}t}} \gamma_2^2 B \times 1 \times \mathrm{d}\mu_{\mathcal{K}_{pp}}(\mathrm{d}\sigma^2)$$

$$\gtrsim \gamma_2^2 B \overline{(\sqrt{\gamma_2 B}t)}^{-2+\frac{1}{2\alpha}}$$

$$\gtrsim \bar{\bar{\mathcal{K}}}_{pp}(t, 0).$$

The above directly brings that $[\mathcal{F} * \mathcal{K}](t) \gtrsim \left( \int_0^t \mathcal{F}(s) \, ds \right) \bar{\bar{\mathcal{K}}}_{pp}(t, 0)$ which concludes. $\qquad \square$

Similarly, we now state an upper-bound result on the convolution between the forcing function and the kernel function.

**Proposition H.22.** *Let $\alpha, \beta$ with $\alpha > \frac{1}{4}$ $\alpha, \beta \neq \frac{1}{2}$, $2\alpha + 2\beta > 1$, $\alpha + 1 > \beta$. Then for any $M > 0$, there exists some $C > 0$ such that for any $t \geq 0$, if $\max\{\gamma_2 Bt, (\sqrt{\gamma_3 \bar{B}} t)^2\} \leq M d^{2\alpha}$ and $\gamma_2 Bt \geq 1$,*

$$[\mathcal{F} * \bar{\bar{\mathcal{K}}}_{pp}](t) \leq C \left( \frac{1}{\gamma_2 B} \bar{\bar{\mathcal{K}}}_{pp}(t, 0) + \mathcal{F}(t) \right).$$

*Proof.* Indeed, we distinguish two cases.

*1st case $t \lesssim \frac{\gamma_2}{\gamma_3}$,* then we write

$$\left[ \bar{\bar{\mathcal{K}}}_{pp} * \mathcal{F} \right](t) \lesssim \int_0^t \gamma_2^2 B \overline{\gamma_2 B(t - s)}^{-2 + 1/(2\alpha)} \mathcal{F}(s) \, ds$$

$$\lesssim \int_0^{t/2} \gamma_2^2 B \overline{(\gamma_2 B(t - s)}^{-2 + 1/(2\alpha)} \mathcal{F}(s) \, ds + \int_{t/2}^t \gamma_2^2 B \overline{(\gamma_2 B(t - s)}^{-2 + 1/(2\alpha)} \mathcal{F}(s) \, ds$$

$$\lesssim \bar{\bar{\mathcal{K}}}_{pp}(t, 0) \int_0^t \mathcal{F}(s) \, ds + \mathcal{F}(t) \int_0^{\frac{\gamma_2}{\gamma_3}} \bar{\bar{\mathcal{K}}}_{pp}(t, s) \, ds$$

$$\lesssim \frac{1}{\gamma_2 B} \bar{\bar{\mathcal{K}}}_{pp}(t, 0) + \mathcal{F}(t).$$

Here to bound the forcing function norm, we used that either $2\beta > 1$ and $\frac{\gamma_2}{\gamma_3} \lesssim \frac{d^{2\alpha}}{\gamma_2 B}$ $\iff$ $\left( \frac{\sqrt{\gamma_3 B}}{\gamma_2 B} \right)^2 \gtrsim d^{-2\alpha}$

$$\int_0^{\frac{\gamma_2}{\gamma_3}} \mathcal{F}(s) \, ds \lesssim \int_0^{\frac{\gamma_2}{\gamma_3}} \mathcal{F}_0(s) \, ds + \int_0^{\frac{\gamma_2}{\gamma_3}} \mathcal{F}_{pp}(s) \, ds + \int_0^{\frac{\gamma_2}{\gamma_3}} \mathcal{F}_{ac}(s) \, ds$$

$$\lesssim d^{-2\alpha} \times \frac{d^{2\alpha}}{\gamma_2 B} + \frac{1}{\gamma_2 B} + \mathbb{1}_{2\alpha > 1} d^{-1} \frac{1}{\gamma_2 B} \left( 1 + \left( \frac{\gamma_2}{\gamma_3} \right)^{1/2\alpha} \right)$$

$$\lesssim \frac{1}{\gamma_2 B}.$$

When $2\beta < 1$ we do not need to worry about $\mathcal{F}_{ac}$ and we write for the $\mathcal{F}_0$ term

$$\bar{\bar{\mathcal{K}}}_{pp}(t, 0) \int_0^t \mathcal{F}_0(s) \, ds \lesssim \gamma_2^2 B \overline{\gamma_2 Bt}^{-2 + \frac{1}{2\alpha}} t d^{-2\alpha - 2\beta + 1}$$

$$\lesssim \gamma_2 \overline{\gamma_2 Bt}^{-1 + \frac{1}{2\alpha}} d^{-2\alpha - 2\beta + 1}$$

$$\lesssim \overline{\gamma_2 Bt}^{-1 - \frac{2\beta - 1}{2\alpha}}$$

$$\lesssim \mathcal{F}(t)$$

where we used $\gamma_2 \lesssim d^{-(1 - 2\alpha)_+}$ and $\gamma_2 Bt \lesssim d^{2\alpha}$.

Finally, for the pure-point term we write

$$\bar{\bar{\mathcal{K}}}_{pp}(t, 0) \int_0^t \mathcal{F}_{pp}(s) \, ds \lesssim \gamma_2^2 B \overline{\gamma_2 Bt}^{-2 + \frac{1}{2\alpha}} \left( \frac{1}{\gamma_2 B} + \frac{1}{\gamma_2 B} (\gamma_2 Bt)^{-\frac{2\beta - 1}{2\alpha}} \right)$$

$$\lesssim \frac{1}{\gamma_2 B} \bar{\bar{\mathcal{K}}}_{pp}(t, 0) + \gamma_2 \overline{\gamma_2 Bt}^{-2 - \frac{\beta - 1}{\alpha}}$$

$$\lesssim \frac{1}{\gamma_2 B} \bar{\bar{\mathcal{K}}}_{pp}(t, 0) + \mathcal{F}(t)$$

where we used that $\gamma_2 \lesssim d^{-(1-2\alpha)_+} \lesssim \overline{\gamma_2 B t}^{1-\frac{1}{2\alpha}}$.

*2nd case:* $t \geq 2\frac{\gamma_2}{\gamma_3}$, we write

$$
\left[ \bar{\bar{\mathcal{K}}}_{pp} * \mathcal{F} \right](t) \lesssim \int_{s=0}^{\frac{\gamma_2}{\gamma_3}} \gamma_2^2 B \overline{\sqrt{\gamma_3 B}(1+t)}^{-4+1/\alpha} \mathcal{F}(s)\, \mathrm{d}s
$$

$$
+ \int_{s=\frac{\gamma_2}{\gamma_3}}^{t-\frac{\gamma_2}{\gamma_3}} \left( \gamma_3 \left( \frac{1+t}{1+s} \right)^{-\delta} (\gamma_2 B(t-s))^{-1+1/(2\alpha)} \right.
$$

$$
\left. + \gamma_2^2 B (\sqrt{\gamma_3 B}(1+t))^{-4+1/\alpha} \left( \frac{1+s}{\frac{\gamma_2}{\gamma_3}} \right)^2 \right) \mathcal{F}(s)\, \mathrm{d}s
$$

$$
+ \int_{s=t-\frac{\gamma_2}{\gamma_3}}^{t} \gamma_2^2 B \overline{\gamma_2 B(t-s)}^{-2+1/(2\alpha)} \mathcal{F}(s)\, \mathrm{d}s
$$

$$
\lesssim \mathcal{F}(t) + \frac{1}{\gamma_2 B} \bar{\bar{\mathcal{K}}}_{pp}(t,0).
$$

Indeed, we used for the first term the same arguments than in the first case to directly obtain

$$
\int_{s=0}^{\frac{\gamma_2}{\gamma_3}} \gamma_2^2 B (\sqrt{\gamma_3 B}(1+t))^{-4+1/\alpha} \mathcal{F}(s)\, \mathrm{d}s \lesssim \frac{1}{\gamma_2 B} \bar{\bar{\mathcal{K}}}_{pp}(t,0) + \mathcal{F}(t).
$$

For the last term note that

$$
\int_{s=t-\frac{\gamma_2}{\gamma_3}}^{t} \gamma_2^2 B \overline{(\gamma_2 B(t-s)}^{-2+1/(2\alpha)} \mathcal{F}(s)\, \mathrm{d}s \lesssim \left( \int_{t-\frac{\gamma_2}{\gamma_3}}^{t} \bar{\bar{\mathcal{K}}}_{pp}(t,s)\, \mathrm{d}s \right) \mathcal{F}(t) \lesssim \mathcal{F}(t).
$$

The middle term needs more care. For the first part, we used that $\delta > \max\{1, 4 - \frac{1}{\alpha}, 2 + \frac{2\beta-1}{\alpha}\}$. This ensures that $\mathcal{F}(s)(1+s)^\delta$ diverges as a power law. Hence we can write

$$
\int_{s=\frac{\gamma_2}{\gamma_3}}^{t-\frac{\gamma_2}{\gamma_3}} \gamma_3 \left( \frac{1+t}{1+s} \right)^{-\delta} (\gamma_2 B(t-s))^{-1+1/(2\alpha)} \mathcal{F}(s)\, \mathrm{d}s
$$

$$
\lesssim \gamma_3 (1+t)^{-\delta} \left[ \mathcal{F}(t)(1+t)^\delta \times t \right] (\gamma_2 B t)^{-1+1/(2\alpha)}
$$

$$
\lesssim \mathcal{F}(t) \times \frac{\gamma_3}{\gamma_2 B} \times (\gamma_2 B t)^{1/(2\alpha)} + \frac{1}{\gamma_2 B} \bar{\bar{\mathcal{K}}}_{pp}(t,0)
$$

$$
\lesssim \mathcal{F}(t) + \frac{1}{\gamma_2 B} \bar{\bar{\mathcal{K}}}_{pp}(t,0)
$$

where we used

$$
\int_{s=\frac{\gamma_2}{\gamma_3}}^{t-\frac{\gamma_2}{\gamma_3}} \mathcal{F}(s)(1+s)^\delta\, \mathrm{d}s \lesssim \mathcal{F}(t)(1+t)^\delta \times t.
$$

We additionally used the **stability condition** $\frac{\gamma_3}{\gamma_2 B} \lesssim d^{-1} \lesssim (\gamma_2 B t)^{\frac{1}{2\alpha}}$.

For the other part we write

$$
\int_{s=\frac{\gamma_2}{\gamma_3}}^{t-\frac{\gamma_2}{\gamma_3}} \gamma_2^2 B (\sqrt{\gamma_3 B}(1+t))^{-4+1/\alpha} \left( \frac{1+s}{\frac{\gamma_2}{\gamma_3}} \right)^2 \mathcal{F}(s)\, \mathrm{d}s
$$

$$
\lesssim \left( \mathcal{F}(t) \times \frac{t^3}{\left( \frac{\gamma_2}{\gamma_3} \right)^2} + \mathcal{F}(\frac{\gamma_2}{\gamma_3}) \right) \times \gamma_2^2 B (\sqrt{\gamma_3 B}(1+t))^{-4+1/\alpha}
$$

$$
\lesssim \mathcal{F}(t) \times \sqrt{\gamma_3 B} \times 1/B (\sqrt{\gamma_3}(1+t))^{-1+1/\alpha} + \mathcal{F}(\frac{\gamma_2}{\gamma_3}) \times \bar{\bar{\mathcal{K}}}_{pp}(t,0)
$$

$$
\lesssim \mathcal{F}(t) + \frac{1}{\gamma_2 B} \bar{\bar{\mathcal{K}}}_{pp}(t,0).
$$

Here we wrote that (using $\frac{\gamma_2 B}{\sqrt{\gamma_3 B}} \gtrsim 1$ for the pure-point and absolutely continuous-part terms)

$$\int_{s=\frac{\gamma_2}{\gamma_3}}^{t-\frac{\gamma_2}{\gamma_3}} \mathcal{F}(s)(1+s)^2 \, \mathrm{d}s \lesssim \mathcal{F}(t)(1+t)^3 + \mathcal{F}(\frac{\gamma_2}{\gamma_3}) \times \left(\frac{\gamma_2}{\gamma_3}\right)^2.$$

We additionally used in the last step for the first term that either $\alpha \leq 1$ and $\sqrt{\gamma_3 B}(1+t) \leq d^\alpha$ and $\sqrt{\gamma_3} = d^{-1/2-(1-2\alpha)_+/2}$, or $\alpha > 1$ and $\sqrt{\gamma_3 B}t \geq \frac{\gamma_2 B}{\sqrt{\gamma_3 B}} \geq 1$ and $\gamma_3 \leq 1$ to obtain that

$$\mathcal{F}(t)\frac{t^3 \gamma_3^2}{\gamma_2^2} \times \gamma_2^2 B(\sqrt{\gamma_3 B}(1+t))^{-4+\frac{1}{\alpha}} \lesssim \mathcal{F}(t)\sqrt{\frac{\gamma_3}{B}}(\sqrt{\gamma_3 B}(1+t))^{-1+1/\alpha} \lesssim \mathcal{F}(t).$$

For the second term, we used $\mathcal{F}(\frac{\gamma_2}{\gamma_3}) \lesssim 1 \lesssim \frac{1}{\gamma_2 B}$.

$\square$

We can now state the main theorem for bounding the loss of DANA-constant.

**Theorem H.3** (Bounds for stable algorithm). *Let $\alpha, \beta \neq \frac{1}{2}$, $\alpha > \frac{1}{4}$, $2\alpha + 2\beta > 1$, $\alpha + 1 > \beta$. Suppose $2\delta \notin \mathbb{N}$, $\delta > \max\{1, 4 - \frac{1}{\alpha}, 2 + \frac{2\beta-1}{\alpha}, 2 - \frac{1}{\alpha}\}$. Under Parametrization H.1, there exists some $c > 0$ such that if $\kappa_1 \geq (1-2\alpha)_+$, $\kappa_2 \geq \kappa_1 - \kappa_b + 1$, $\kappa_b \leq \min\{\kappa_1, \kappa_2\}$, $\tilde{\gamma}_2 \leq c$, $\frac{\tilde{\gamma}_3}{\tilde{\gamma}_2} \leq c$, then $\mathcal{P}(t)$ is bounded and there exists an $M > 0$ large enough and a constant $\tilde{C}(\alpha, \beta, M)$ such that if $Md^{2\alpha} > \gamma_2 Bt, (\sqrt{\gamma_3 B}(1+t))^2 > 1$ then:*

$$\frac{1}{\tilde{C}} \times \left(\mathcal{F}_0(t) + \mathcal{F}_{ac}(t) + \mathcal{F}_{pp}(t) + \frac{1}{\gamma_2 B}\bar{\bar{\mathcal{K}}}_{pp}(t,0)\right) \leq \mathcal{P}(t)$$

$$\leq \tilde{C} \times \left(\mathcal{F}_0(t) + \mathcal{F}_{ac}(t) + \mathcal{F}_{pp}(t) + \frac{1}{\gamma_2 B}\bar{\bar{\mathcal{K}}}_{pp}(t,0)\right).$$

*Proof.* **Upper-bound**

We first apply Lemma C.2. Using the bounds on the kernel norm and forcing function convergence guarantees for $c$ small enough that if $\kappa_1 \geq (1-2\alpha)_+$, $\kappa_2 \geq \kappa_1 - \kappa_b + 1$, $\kappa_b \leq \min\{\kappa_1, \kappa_2\}$, $\tilde{\gamma}_2 \leq c$, $\frac{\tilde{\gamma}_3}{\tilde{\gamma}_2} \leq c$ then,

$$\mathcal{P}(t) \leq \tilde{C} \times \left(\mathcal{F}(t) + [\bar{\bar{\mathcal{K}}}_{pp} * \mathcal{F}](t)\right).$$

We only need to estimate $[\bar{\bar{\mathcal{K}}}_{pp} * \mathcal{F}](t)$ which was done in Proposition H.22, and obtain for a potentially different $\tilde{c}, \tilde{C}$,

$$[\bar{\bar{\mathcal{K}}}_{pp} * \mathcal{F}](t) \leq \tilde{C} \times \left(\mathcal{F}(t) + \frac{1}{\gamma_2 B}\bar{\bar{\mathcal{K}}}_{pp}(t,0)\right).$$

**Lower bound**

We know that $\forall t \geq 0, \mathcal{P}(t) \geq \mathcal{F}(t) + [\mathcal{F} * \mathcal{K}](t)$ and hence only need to lower bound $[\mathcal{F} * \mathcal{K}](t)$. This was done in Proposition H.21.

$\square$

# I   DANA-decaying

Below we introduce the main parametrization for DANA-decaying that will be used throughout this Section I.

**Parametrization I.1.** *For $\gamma_2$, $c$ $\kappa$ constants*

$$\gamma_1 = 1, \quad \gamma_2(t) \equiv \gamma_2, \quad \gamma_3(t) \stackrel{def}{=} \gamma_3(1+t)^{-\kappa} \stackrel{def}{=} \gamma_2 c(1+t)^{-\kappa}. \tag{108}$$

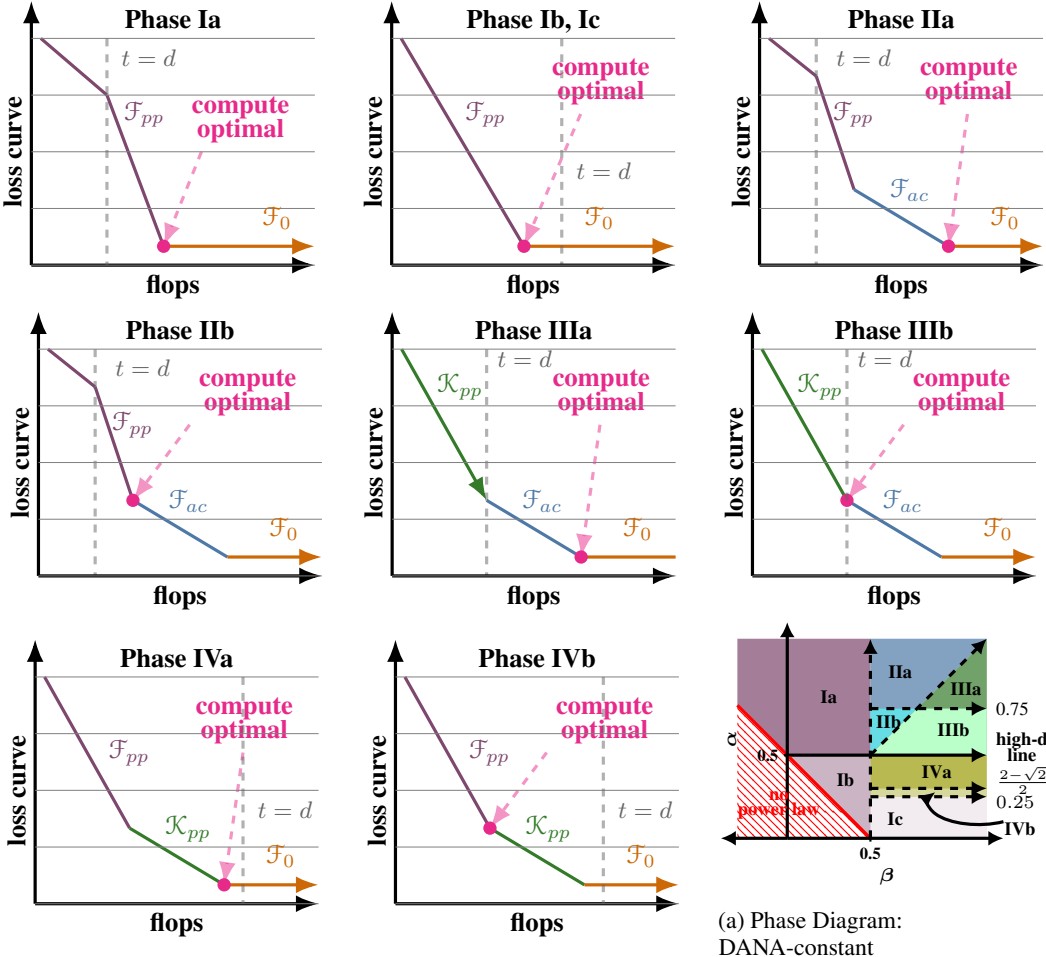

Figure 18: **Cartoon Plots DANA-constant.** Pictures for the scaling laws for DANA-constant in each of the different phases. When $t < d$, DANA-constant behaves like SGD/SGD-M. Observe that the trade off point for compute-optimum changes across phases (see Sec. E for derivations and Table 6). See Table 10 for summary of asymptotics of $\mathcal{F}_{pp}, \mathcal{F}_{ac}, \mathcal{F}_0$, and $\mathcal{K}_{pp}$. This uses DANA-constant with $\kappa_2 = 1$ and batch size $B = 1$.

**Assumption 6.** *In all this section, we suppose $1 > \kappa > \frac{1}{2\alpha}$ with $d$-independent constant learning rates $\gamma_2 \asymp \gamma_3 \asymp 1$, and $B = 1$. Moreover, we only work above the high-dimensional line $2\alpha > 1$ and under the technical assumption $\beta < \alpha + 1$. Finally we suppose $2\alpha + 2\beta > 1$ and $\beta \neq \frac{1}{2}$. We also suppose $\delta$ large enough (independent of $d$).*

We will later heuristically extend the results in this section under this assumption to the general (DANA) algorithm in Section I.5.

**Remark I.1.** *We supposed $\alpha > \frac{1}{2}$ as it will become clear that for $\alpha < \frac{1}{2}$, this algorithm is equivalent to SGD. Indeed in that case $\vartheta(t) \asymp 1 + \gamma_2 Bt$. The scaling $\gamma_2 \asymp \gamma_3 \asymp 1$ and $1 \geq \kappa > \frac{1}{2\alpha}$ will imply the relation between momentum and SGD times for eigenvalue $\sigma^2$,*

$$\sqrt{\gamma_3(t)B}\sigma(1+t) \gtrsim \sqrt{\gamma_2 B\sigma^2(1+t)}.$$

We also remind the definition $1 + 2\gamma_2 Bt + \left(\int_0^t \sqrt{\gamma_3(s)B}\, ds\right)^2$ that will be used throughout this section.

## I.1 Solutions to the simplified ODE

As before, we work with the simplified ODEs (43) and the resulting forcing function, kernel function and simplified Volterra equation solution (55).

We now provide explicit estimates for the DANA-decaying solution, $\Phi_{\sigma^2}(t,s)$, to the simplified ODEs (43).

**Theorem I.1** (Fundamental solutions). *We summarize the results of this section below. Suppose that* $\delta + \kappa > 1$ *and suppose that* $\mathfrak{m}^{1-\kappa} \le c \le 1$. *Set for convenience*

$$\mathfrak{s}(t) = c(1+t)^{-\kappa}, \quad and \quad \mathfrak{m} = \gamma_2 B \sigma^2. \tag{109}$$

*where* $\kappa \in (0,1)$ *is constant.*

*There is a* $\mathfrak{c} > 0$ *so that for all* $\epsilon > 0$ *there is an* $\mathfrak{m}_0$ *sufficiently small so that for all* $\mathfrak{m} < \mathfrak{m}_0$, *the following hold.*

*We use the time scale* $\tau = \mathfrak{m}(1+t)$ *and*

$$\xi(\tau;\tau_0) = \int_{\tau_0}^{\tau} \sqrt{\frac{\mathfrak{s}}{\mathfrak{m}}}\, du = \sqrt{c}\frac{\tau^{1-\kappa/2} - \tau_0^{1-\kappa/2}}{1-\kappa/2} \mathfrak{m}^{-1/2+\kappa/2}.$$

*We let* $\xi(\tau) = \xi(\tau;\mathfrak{m})$ *for short.*

*There are* $\omega_1(\xi)$ *and* $\omega_2(\xi)$ *continuously differentiable functions with* $\omega_1^2 + \omega_2^2$ *bounded away from* $0$ *and above by a constant, and they satisfy*

$$\begin{pmatrix}\omega_1(\xi)\\\omega_2(\xi)\end{pmatrix} \to \begin{pmatrix}1\\0\end{pmatrix} \quad as\ \xi \to 0 \quad and \quad \begin{pmatrix}\omega_1(\xi)\\\omega_2(\xi)\end{pmatrix} = \sqrt{\frac{2}{\pi}}\begin{pmatrix}\cos(\xi + \omega_\nu(0))\\\sin(\xi + \omega_\nu(0))\end{pmatrix} + O(\xi^{-1})\ as\ \xi \to \infty,$$

*for some constant* $\omega_\nu(0)$.

*For any* $s \ge 0$, *we use* $\tau_0 = \mathfrak{m}(1+s)$ *and* $\xi_0$ *the corresponding value of* $\xi$. *Then with* $\rho = \frac{\delta/2+\kappa/4}{1-\kappa/2}$, *we have the following uniform estimate for* $\xi_0 < \epsilon$

$$\begin{pmatrix}\Phi_{11}(t;s)\\\frac{(\delta+\kappa-1)^2}{((1-\kappa/2)\xi(\tau_0;0))^2}\frac{\mathfrak{m}}{\gamma_2^2\mathfrak{s}(s)}\Phi_{12}(t;s)\end{pmatrix} = e^{-\tau}(1+\xi)^{-2\rho}\begin{pmatrix}\omega_1(\xi)^2 + O(\epsilon)\\\omega_1(\xi)^2(1-s/t)^2 + O(\epsilon)\end{pmatrix}$$
$$+ O\left(\epsilon\mathfrak{m}^{\delta+\kappa/2}\Omega^2(\tau)e^{-\int_0^\tau \Omega(u)\,du}\right) \quad where \quad \Omega(u) = \min\{1, (\xi'(u))^2\}. \tag{110}$$

*The error term* $O(\epsilon)$ *also vanishes quadratically in the second entry as* $\xi \to 0$.

*For all* $\tau_0 < \frac{1}{\epsilon}$ *with* $s$ *large enough that* $\xi_0 > \epsilon$ *there are other bounded oscillatory* $\omega_1, \omega_2$

$$\begin{pmatrix}\Phi_{11}(t;s)\\\frac{\mathfrak{m}}{\gamma_2^2\mathfrak{s}(s)}\Phi_{12}(t;s)\end{pmatrix} = e^{-(\tau-\tau_0)}\left(\frac{1+\xi}{1+\xi_0}\right)^{-2\rho}\begin{pmatrix}\omega_1(\xi;\xi_0)^2 + O(\epsilon)\\\omega_2(\xi;\xi_0)^2 + O(\epsilon)\end{pmatrix}$$
$$+ O\left(\epsilon\mathfrak{m}^{\delta+\kappa/2}\Omega^2(\tau)e^{-\int_{\tau_0}^\tau \Omega(u)\,du}\right).$$

*If* $\tau_0 > \frac{1}{\epsilon}$

$$\begin{pmatrix}\Phi_{11}(t;s)\\\frac{\mathfrak{m}}{\gamma_2^2\mathfrak{s}(s)}\Phi_{12}(t;s)\end{pmatrix} = \begin{pmatrix}e^{-(\tau-\tau_0)}\\0\end{pmatrix} + O\left(\Omega^2(\tau)e^{-\int_{\tau_0}^\tau \Omega(u)\,du}\right).$$

We summarize the regimes for this fundamental solution below. In the first training regime, when $t$ is small ($\xi \ll 1$), which is to say

$$\xi \asymp \mathfrak{m}^{1/2}(1+t)^{1-\kappa/2} \ll 1,$$

we have that the ODEs have not begun moving. On the time scale from $\xi \gg 1$ but $\tau \ll 1$ we are not yet using the curvature, but we have decay like

$$\xi^{-2\rho} \asymp \mathfrak{m}^{-\rho}(1+t)^{-\delta-\kappa/2}.$$

Finally once $t \gg \mathfrak{m}^{-1}$, there is exponential decay with speed $\mathfrak{m}$.

In the remainder of this section, we prove this result.

**Changing variables.**  We recall (99)

$$\frac{\mathrm{d}\Phi_{\sigma^2}(t)}{\mathrm{d}t} = \left( \frac{\begin{pmatrix} 0 & 0 & 0 \\ 0 & -2\delta & 0 \\ 0 & 0 & -\delta \end{pmatrix}}{1+t} + \begin{pmatrix} -2\gamma_2 B\sigma^2 & 0 & -2\gamma_3(t) \\ 0 & 0 & 2\gamma_1 B\sigma^2 \\ \gamma_1 B\sigma^2 & -\gamma_3(t) & -\gamma_2 B\sigma^2 \end{pmatrix} \right) \Phi_{\sigma^2}(t).$$

To simplify the computations, we will additionally do the change of variable $\tilde{\Phi}(t) \overset{\text{def}}{=} \begin{pmatrix} \Phi_1(t) \\ \frac{1}{B^2\sigma^4}\Phi_2(t) \\ \frac{1}{B\sigma^2}\Phi_3(t) \end{pmatrix}$

on Equation (99) and get the new ODE

$$\frac{\mathrm{d}\tilde{\Phi}(t)}{\mathrm{d}t} = \left( \frac{1}{1+t}\begin{pmatrix} 0 & 0 & 0 \\ 0 & -2\delta & 0 \\ 0 & 0 & -\delta \end{pmatrix} + \begin{pmatrix} -2\gamma_2 B\sigma^2 & 0 & -2\gamma_3 B\sigma^2 \\ 0 & 0 & 2 \\ 1 & -\gamma_3 B\sigma^2 & -\gamma_2 B\sigma^2 \end{pmatrix} \right) \tilde{\Phi}(t). \quad (111)$$

In terms of the variables $\mathfrak{m}$ and $\mathfrak{s}$ we can rewrite the simplified ODE (111) as

$$\frac{\mathrm{d}\tilde{\Phi}(t)}{\mathrm{d}t} = \left( \frac{1}{1+t}\begin{pmatrix} 0 & 0 & 0 \\ 0 & -2\delta & 0 \\ 0 & 0 & -\delta \end{pmatrix} + \begin{pmatrix} -2\mathfrak{m} & 0 & -2\mathfrak{m}\mathfrak{s} \\ 0 & 0 & 2 \\ 1 & -\mathfrak{m}\mathfrak{s} & -\mathfrak{m} \end{pmatrix} \right) \tilde{\Phi}(t). \quad (112)$$

We now introduce a time change $\tau(t) \overset{\text{def}}{=} \mathfrak{m}(1+t)$ and defining $\hat{\Phi}(\tau) \overset{\text{def}}{=} \tilde{\Phi}(t)$, in terms of which we obtain the new ODE:

$$\frac{\mathrm{d}\hat{\Phi}(\tau)}{\mathrm{d}\tau} \overset{\text{def}}{=} \left( \frac{R}{\tau} + A \right) \hat{\Phi}(\tau) = \left( \frac{1}{\tau}\begin{pmatrix} 0 & 0 & 0 \\ 0 & -2\delta & 0 \\ 0 & 0 & -\delta \end{pmatrix} + \begin{pmatrix} -2 & 0 & -2\mathfrak{s} \\ 0 & 0 & \frac{2}{\mathfrak{m}} \\ \frac{1}{\mathfrak{m}} & -\mathfrak{s} & -1 \end{pmatrix} \right) \hat{\Phi}(\tau). \quad (113)$$

**Lemma I.1.** *Suppose that $X$ and $Y$ solve the ODE*

$$\frac{\mathrm{d}}{\mathrm{d}\tau}\begin{pmatrix} X \\ Y \end{pmatrix} = \begin{pmatrix} -1 & -\mathfrak{s} \\ \frac{1}{\mathfrak{m}} & -\frac{\delta}{\tau} \end{pmatrix}\begin{pmatrix} X \\ Y \end{pmatrix}. \quad (114)$$

*Then the vector $\left(X^2, Y^2, XY\right)$ solves (113). Hence, a fundamental matrix for (113) can be given in terms of solutions to (114) with initial data at time $\tau_0$ given by*

$$\begin{aligned}
(X^2, Y^2, XY) \quad &\text{where} \quad (X(\tau_0), Y(\tau_0)) = (1, 0), \\
(X^2, Y^2, XY) \quad &\text{where} \quad (X(\tau_0), Y(\tau_0)) = (0, 1/(B\sigma^2)), \\
\mathrm{Im}(X^2, Y^2, XY) \quad &\text{where} \quad (X(\tau_0), Y(\tau_0)) = (1, i/(B\sigma^2)).
\end{aligned}$$

*Proof.* We just need to verify that the equation is satisfied by the vector $\left(X^2, Y^2, XY\right)$.

$$\frac{\mathrm{d}}{\mathrm{d}\tau}\begin{pmatrix} \Phi_1 \\ \Phi_2 \\ \Phi_3 \end{pmatrix} = \frac{\mathrm{d}}{\mathrm{d}\tau}\begin{pmatrix} X^2 \\ Y^2 \\ XY \end{pmatrix} = \begin{pmatrix} 2X\frac{\mathrm{d}X}{\mathrm{d}\tau} \\ 2Y\frac{\mathrm{d}Y}{\mathrm{d}\tau} \\ X\frac{\mathrm{d}Y}{\mathrm{d}\tau} + Y\frac{\mathrm{d}X}{\mathrm{d}\tau} \end{pmatrix} = \begin{pmatrix} 2X(-1X - \mathfrak{s}Y) \\ 2Y(\frac{1}{\mathfrak{m}}X - \frac{\delta}{\tau}Y) \\ X(\frac{1}{\mathfrak{m}}X - \frac{\delta}{\tau}Y) + Y(-1X - \mathfrak{s}Y) \end{pmatrix}$$

$$= \begin{pmatrix} -2\Phi_1 - 2\mathfrak{s}\Phi_3 \\ \frac{2}{\mathfrak{m}}\Phi_3 - 2\frac{\delta}{\tau}\Phi_2 \\ \frac{1}{\mathfrak{m}}\Phi_1 - \mathfrak{s}\Phi_2 - (1 + \frac{\delta}{\tau})\Phi_3 \end{pmatrix}.$$

This is precisely the equation (113). □

To formalize the estimates, we divide the range of $\tau$ into three regimes. Set

$$\mathfrak{p} = \frac{1/2 - \kappa/2}{1 - \kappa/2} < 1. \quad (115)$$

- (Entrance) $\tau < \epsilon \mathfrak{m}^{\mathfrak{p}} : (\xi \ll 1)$ Here we will approximate $X$ and $Y$ by simple power expressions.

- (Transition) $\epsilon \mathfrak{m}^{\mathfrak{p}} < \tau < \frac{1}{\epsilon} \mathfrak{m}^{\mathfrak{p}} : (\xi \asymp 1)$ Here the fundamental matrix solves a non-degenerate rescaled ODE, which after changing variables is an approximate solution of the Bessel equation.

- (Bulk) $\frac{1}{\epsilon} \mathfrak{m}^{\mathfrak{p}} < \tau < \frac{1}{\epsilon} : (\xi \gg 1$ and $\tau \ll 1)$ Here the system develops a very strong oscillatory behavior which we can describe explicitly.

- (Exponential decay) $\frac{1}{\epsilon} \leq \tau \leq \epsilon \mathfrak{m}^{1-\frac{1}{\kappa}} : (\tau \gg 1$ and $\xi'(\tau) \gg 1)$ Here we will give estimates showing uniform exponential decay of the fundamental matrix.

- (Slow decay) $\epsilon \mathfrak{m}^{1-\frac{1}{\kappa}} < \tau : (\xi'(\tau) \ll 1)$ Here the fundamental matrix continues to decay, but at a slower stretched-exponential rate.

**First change of variables: Isotropic coordinates**   We start by making a change of variables. Introduce $\mathcal{Y} = \sqrt{\mathfrak{sm}}Y$. Then, changing variables from (114) to $\mathcal{Y}$, we get

$$
\begin{aligned}
\frac{\mathrm{d}\mathcal{Y}}{\mathrm{d}\tau} &= \frac{\mathrm{d}}{\mathrm{d}\tau}\left(\sqrt{\mathfrak{sm}}Y\right) \\
&= \frac{\mathrm{d}}{\mathrm{d}\tau}\left(\sqrt{\mathfrak{sm}}\right)Y + \sqrt{\mathfrak{sm}}\frac{\mathrm{d}Y}{\mathrm{d}\tau} \\
&= \frac{1}{2}\frac{\mathrm{d}\mathfrak{s}}{\mathrm{d}\tau}\frac{1}{\mathfrak{s}}\sqrt{\mathfrak{sm}}Y + \sqrt{\mathfrak{sm}}\left(\frac{1}{\mathfrak{m}}X - \frac{\delta}{\tau}Y\right)
\end{aligned}
$$

Since $\mathfrak{s}(t) = c(1+t)^{-\kappa}$ and $\tau = \mathfrak{m}(1+t)$, we have $\frac{\mathrm{d}\mathfrak{s}}{\mathrm{d}\tau} = -\kappa \frac{\mathfrak{s}}{\tau}$. Thus:

$$
\begin{aligned}
\frac{\mathrm{d}\mathcal{Y}}{\mathrm{d}\tau} &= -\frac{\kappa}{2\tau}\mathcal{Y} + \sqrt{\mathfrak{sm}}\frac{1}{\mathfrak{m}}X - \frac{\delta}{\tau}\mathcal{Y} \\
&= \sqrt{\mathfrak{sm}}\frac{1}{\mathfrak{m}}X - \left(\frac{\delta + \kappa/2}{\tau}\right)\mathcal{Y}.
\end{aligned}
$$

Therefore, our system becomes:

$$
\frac{\mathrm{d}}{\mathrm{d}\tau}\begin{pmatrix} X \\ \mathcal{Y} \end{pmatrix} = \begin{pmatrix} -1 & -\sqrt{\frac{\mathfrak{s}}{\mathfrak{m}}} \\ \sqrt{\frac{\mathfrak{s}}{\mathfrak{m}}} & -\frac{\delta + \kappa/2}{\tau} \end{pmatrix}\begin{pmatrix} X \\ \mathcal{Y} \end{pmatrix}. \tag{116}
$$

We define fundamental matrices of this matrix equation

$$
\frac{\mathrm{d}}{\mathrm{d}\tau}\mathcal{P}(\tau;\tau_0) = \begin{pmatrix} -1 & -\sqrt{\frac{\mathfrak{s}}{\mathfrak{m}}} \\ \sqrt{\frac{\mathfrak{s}}{\mathfrak{m}}} & -\frac{\delta + \kappa/2}{\tau} \end{pmatrix}\mathcal{P}(\tau;\tau_0), \quad \mathcal{P}(\tau_0;\tau_0) = \mathrm{Id}. \tag{117}
$$

**Second change of variables: Rotating frame**   Define the orthogonal matrix $\mathcal{R}(\tau)$ by

$$
\mathcal{R}(\tau;\tau_0) = \begin{pmatrix} \cos(\xi) & -\sin(\xi) \\ \sin(\xi) & \cos(\xi) \end{pmatrix}, \quad \xi(\tau;\tau_0) = \int_{\tau_0}^{\tau}\sqrt{\frac{\mathfrak{s}(u)}{\mathfrak{m}}}\,\mathrm{d}u. \tag{118}
$$

We observe that the matrix $\mathcal{R}$ satisfies the following ODE:

$$
\frac{\mathrm{d}}{\mathrm{d}\tau}\mathcal{R} = \begin{pmatrix} -\sin(\xi) & -\cos(\xi) \\ \cos(\xi) & -\sin(\xi) \end{pmatrix}\xi' = \xi'\begin{pmatrix} 0 & -1 \\ 1 & 0 \end{pmatrix}\mathcal{R}. \tag{119}
$$

Thus if we set

$$
\begin{pmatrix} U \\ V \end{pmatrix} = \mathcal{R}^{-1}\begin{pmatrix} X \\ \mathcal{Y} \end{pmatrix},
$$

then differentiating with respect to $\tau$, we get

$$\frac{\mathrm{d}}{\mathrm{d}\tau}\begin{pmatrix} U \\ V \end{pmatrix} = \frac{\mathrm{d}}{\mathrm{d}\tau}\left( \mathcal{R}^{-1}\begin{pmatrix} X \\ \mathcal{Y} \end{pmatrix} \right)$$

$$= \frac{\mathrm{d}\mathcal{R}^{-1}}{\mathrm{d}\tau}\begin{pmatrix} X \\ \mathcal{Y} \end{pmatrix} + \mathcal{R}^{-1}\frac{\mathrm{d}}{\mathrm{d}\tau}\begin{pmatrix} X \\ \mathcal{Y} \end{pmatrix}$$

$$= -\mathcal{R}^{-1}\frac{\mathrm{d}\mathcal{R}}{\mathrm{d}\tau}\mathcal{R}^{-1}\begin{pmatrix} X \\ \mathcal{Y} \end{pmatrix} + \mathcal{R}^{-1}\frac{\mathrm{d}}{\mathrm{d}\tau}\begin{pmatrix} X \\ \mathcal{Y} \end{pmatrix}.$$

Using equation (119) for $\frac{\mathrm{d}\mathcal{R}}{\mathrm{d}\tau}$ and equation (116) for $\frac{\mathrm{d}}{\mathrm{d}\tau}\begin{pmatrix} X \\ \mathcal{Y} \end{pmatrix}$, we obtain:

$$\frac{\mathrm{d}}{\mathrm{d}\tau}\begin{pmatrix} U \\ V \end{pmatrix} = -\mathcal{R}^{-1}\left( \xi'\begin{pmatrix} 0 & -1 \\ 1 & 0 \end{pmatrix}\mathcal{R} \right)\mathcal{R}^{-1}\begin{pmatrix} X \\ \mathcal{Y} \end{pmatrix}$$

$$+ \mathcal{R}^{-1}\begin{pmatrix} -1 & -\sqrt{\frac{\mathfrak{s}}{\mathfrak{m}}} \\ \sqrt{\frac{\mathfrak{s}}{\mathfrak{m}}} & -\frac{\delta+\kappa/2}{\tau} \end{pmatrix}\begin{pmatrix} X \\ \mathcal{Y} \end{pmatrix}$$

$$= -\xi'\mathcal{R}^{-1}\begin{pmatrix} 0 & -1 \\ 1 & 0 \end{pmatrix}\mathcal{R}\begin{pmatrix} U \\ V \end{pmatrix} + \mathcal{R}^{-1}\begin{pmatrix} -1 & -\sqrt{\frac{\mathfrak{s}}{\mathfrak{m}}} \\ \sqrt{\frac{\mathfrak{s}}{\mathfrak{m}}} & -\frac{\delta+\kappa/2}{\tau} \end{pmatrix}\mathcal{R}\begin{pmatrix} U \\ V \end{pmatrix}.$$

As we have chosen $\xi$ such that $\xi' = \sqrt{\frac{\mathfrak{s}}{\mathfrak{m}}}$, then the off-diagonal terms cancel, and we get

$$\frac{\mathrm{d}}{\mathrm{d}\tau}\begin{pmatrix} U \\ V \end{pmatrix} = \mathcal{R}^{-1}\begin{pmatrix} -1 & 0 \\ 0 & -\frac{\delta+\kappa/2}{\tau} \end{pmatrix}\mathcal{R}\begin{pmatrix} U \\ V \end{pmatrix}. \tag{120}$$

We also introduce the fundamental matrix of this system

$$\frac{\mathrm{d}}{\mathrm{d}\tau}\mathcal{U}(\tau;\tau_0) = \mathcal{R}^{-1}\begin{pmatrix} -1 & 0 \\ 0 & -\frac{\delta+\kappa/2}{\tau} \end{pmatrix}\mathcal{R}\mathcal{U}(\tau;\tau_0), \quad \mathcal{U}(\tau_0;\tau_0) = \mathrm{Id}. \tag{121}$$

We also record for convenience the relation between the fundamental matrices that we will need, (with $\tau = \mathfrak{m}(1+t)$ and $\tau_0 = \mathfrak{m}(1+s)$):

$$\mathcal{P}(\tau;\tau_0) = \mathcal{R}(\tau;\tau_0)\mathcal{U}(\tau;\tau_0) \quad \text{and} \quad \begin{pmatrix} \Phi_{11}(t,s) \\ \Phi_{12}(t,s) \end{pmatrix} = \begin{pmatrix} \mathcal{P}_{11}(\tau;\tau_0) \\ \mathcal{P}_{12}(\tau;\tau_0)(\gamma_2\mathfrak{s}(s)) \end{pmatrix}. \tag{122}$$

To verify the second matrix entry, we use Lemma I.1 to show that

$$\Phi_{12}(t,s) = X^2(t) \quad \text{where} \quad X(s) = 0 \quad \text{and} \quad 1/(B\sigma^2) = Y(s) = \mathcal{Y}(s)/\sqrt{\mathfrak{s}(s)\mathfrak{m}}.$$

Thus

$$\Phi_{12}(t,s) = \mathcal{P}_{12}^2(\tau;\tau_0)\frac{\mathfrak{s}(s)\mathfrak{m}}{(B\sigma^2)^2} = \mathcal{P}_{12}^2(\tau;\tau_0)\frac{\gamma_2^2\mathfrak{s}(s)}{\mathfrak{m}}.$$

**Lemma I.2** (Entrance). *Suppose $\delta > 2$. In what follows we use $\xi(\tau) = \xi(\tau;\mathfrak{m})$. There is a $\mathfrak{c} > 0$ so that for all $\epsilon > 0$ sufficiently small, there is an $\mathfrak{m}_0$, so that if $\mathfrak{m} < \mathfrak{m}_0$ and $\mathfrak{m} < \tau_0 < \tau < \epsilon\mathfrak{m}^\mathfrak{p}$ (hence $\xi(\tau) \lesssim \epsilon^{1-\kappa/2}$)*

$$|\mathcal{P}(\tau;\tau_0)_{11} - e^{-(\tau-\tau_0)}| \le \mathfrak{c}e^{-(\tau-\tau_0)}\xi^2(\tau) \quad \text{and} \quad |\mathcal{P}(\tau;\tau_0)_{21} - I_1(\tau;\tau_0)| \le \mathfrak{c}I_1(\tau;\tau_0)\xi^2(\tau),$$

$$\text{where} \quad I_1(\tau;\tau_0) = \int_{\tau_0}^\tau \left(\frac{u}{\tau}\right)^{\delta+\kappa/2}\xi'(u)e^{-(u-\tau_0)}\,\mathrm{d}u,$$

*and we note $I_1(\tau;\tau_0) \le \mathfrak{c}\xi(\tau;\tau_0)$. We also have that*

$$|\mathcal{P}(\tau;\tau_0)_{12} - I_2(\tau;\tau_0)| \le \mathfrak{c}I_2(\tau;\tau_0)\xi^2(\tau) \quad \text{and} \quad |\mathcal{P}(\tau;\tau_0)_{22} - (\tau/\tau_0)^{-\delta-\kappa/2}| \le \mathfrak{c}I_2(\tau;\tau_0)\xi(\tau),$$

$$\text{where} \quad I_2(\tau;\tau_0) = \int_{\tau_0}^\tau \left(\frac{u}{\tau_0}\right)^{-\delta-\kappa/2}\xi'(u)e^{-(\tau-u)}\,\mathrm{d}u.$$

*We have that $I_2(\tau; \tau_0) \leq \mathfrak{c}\xi(\tau_0; 0)$, and that*

$$I_2(\tau; \tau_0) = \frac{1 - \kappa/2}{\delta + \kappa - 1}\xi(\tau_0; 0)(1 + o(1)),$$

*with the error $o(1)$ tending to $0$ as $\tau/\tau_0 \to \infty$ but $\tau \to 0$.*

*Consequently, for $\Phi$, we have the following bounds in this regime:*

$$\Phi_{11}(t, s) \lesssim 1 \quad and \quad \Phi_{12}(t, s) \lesssim c\gamma_2^2(1 + s)^{2(1-\kappa)}.$$

*Proof.* Under the assumptions of the lemma, we have that for all $\tau_0 < \tau < \epsilon\mathfrak{m}^{\mathfrak{p}}$

$$0 \leq \xi(\tau) \leq \frac{1}{1 - \kappa/2}\sqrt{c}\epsilon^{1-\kappa/2} \quad and \quad \xi'(\tau)\tau \leq \sqrt{c}\epsilon^{1-\kappa/2}.$$

**First column.** We start with the first column of $\mathcal{P}(\tau; \tau_0)$. The natural candidate for the solution to the 11 entry is

$$\mathcal{P}(\tau; \tau_0)_{11} = e^{-(\tau - \tau_0)}.$$

So we introduce

$$\mathcal{P}(\tau; \tau_0)_{11} = e^{-(\tau - \tau_0)}(1 + \mathcal{V}(\tau)),$$

where $\mathcal{V}(\tau_0) = 0$. From (116), we have

$$\mathcal{V}'(\tau) = -\xi'(\tau)\mathcal{P}_{21}(\tau; \tau_0).$$

Integrating by parts, we have

$$\mathcal{V}(\tau) = -\int_{\tau_0}^{\tau}\xi'(u)\mathcal{P}_{21}(u; \tau_0)\, du$$

$$= -\xi(\tau)\mathcal{P}_{21}(\tau; \tau_0) + \int_{\tau_0}^{\tau}\xi(u)\left(\xi'(u)e^{-(\tau - \tau_0)}(1 + \mathcal{V}(u)) - \frac{\delta + \kappa/2}{u}\mathcal{P}_{21}(u; \tau_0)\right)\, du.$$

As for $\mathcal{P}_{21}$, we have

$$\mathcal{P}_{21}(\tau; \tau_0) = \int_{\tau_0}^{\tau}\left(\frac{u}{\tau}\right)^{\delta + \kappa/2}\xi'(u)e^{-(u - \tau_0)}(1 + \mathcal{V}(u))\, du.$$

Let $\mathfrak{v}(\tau) = \max_{\tau_0 \leq u \leq \tau}|\mathcal{V}(u)|$. Then we have a bound

$$|\mathcal{P}_{21}(\tau; \tau_0)| \leq (1 + \mathfrak{v}(\tau))\int_{\tau_0}^{\tau}\left(\frac{u}{\tau}\right)^{\delta + \kappa/2}\xi'(u)\, du$$

$$\leq \frac{(1 + \mathfrak{v}(\tau))}{1 + \delta}\left(\sqrt{c}\mathfrak{m}^{-1/2+\kappa/2}\frac{u^{\delta+1}}{\tau^{\delta+\kappa/2}}\right)\Big|_{\tau_0}^{\tau}$$

$$\leq \frac{(1 + \mathfrak{v}(\tau))}{1 + \delta}\left(\sqrt{c}\mathfrak{m}^{-1/2+\kappa/2}\tau^{1-\kappa/2}\right) = \frac{(1 + \mathfrak{v}(\tau))}{1 + \delta}\xi'(\tau)\tau.$$

So we have, substituting these bounds,

$$|\mathcal{V}(\tau)| \leq \frac{(1 + \mathfrak{v}(\tau))}{1 + \delta}\xi(\tau)\xi'(\tau)\tau + \frac{\xi^2(\tau)}{2}(1 + \mathfrak{v}(\tau)) + \frac{\xi^2(\tau)(\delta + \kappa/2)}{2(1 + \delta)}(1 + \mathfrak{v}(\tau)),$$

and hence

$$|\mathcal{V}(\tau)| \leq C(\delta, \kappa)c\epsilon^{2-\kappa}(1 + \mathfrak{v}(\tau)),$$

and so by monotonicity of $\mathfrak{v}$, we have

$$\mathfrak{v}(\tau) \leq \frac{C(\delta, \kappa)c\epsilon^{2-\kappa}}{1 - C(\delta, \kappa)c\epsilon^{2-\kappa}}.$$

This proves the first part of the lemma.

**Second column.** We now turn to the second column of $\mathcal{P}(\tau; \tau_0)$. We again introduce a function $\mathcal{V}(\tau)$ now defined by

$$\mathcal{P}(\tau; \tau_0)_{22} = \left(\frac{\tau}{\tau_0}\right)^{-\delta - \kappa/2} \times (1 + \mathcal{V}(\tau)),$$

with $\mathcal{V}(\tau_0) = 0$. Then changing variables, from (117), we have

$$\mathcal{V}'(\tau) = \xi'(\tau) \left(\frac{\tau}{\tau_0}\right)^{\delta + \kappa/2} \times \mathcal{P}_{12}(\tau; \tau_0). \tag{123}$$

As for $\mathcal{P}_{12}$, we have from (117)

$$\mathcal{P}_{12}(\tau; \tau_0) = -\int_{\tau_0}^{\tau} \left(\frac{u}{\tau_0}\right)^{-\delta - \kappa/2} \xi'(u) e^{-(\tau - u)} (1 + \mathcal{V}(u)) \, du.$$

Let $\mathfrak{v}(\tau) = \max_{\tau_0 \leq u \leq \tau} \left(\left(\frac{u}{\tau_0}\right)^{-\delta - \kappa/2} \times |\mathcal{V}(u)|\right)$. Define

$$I_2(\tau; \tau_0) = \int_{\tau_0}^{\tau} \left(\frac{u}{\tau_0}\right)^{-\delta - \kappa/2} \xi'(u) e^{-(\tau - u)} \, du.$$

Then we have a bound

$$|\mathcal{P}_{12}(\tau; \tau_0) - I_2(\tau; \tau_0)| \leq \mathfrak{v}(\tau) \xi(\tau).$$

So we have, integrating (123) and using the above bound,

$$\left| \mathcal{V}(\tau) - \int_{\tau_0}^{\tau} \left(\frac{u}{\tau_0}\right)^{\delta + \kappa/2} \xi'(u) I(u; \tau_0) \, du \right| \leq \frac{\xi(\tau)^2}{1 + \delta - \kappa} \left(\frac{\tau}{\tau_0}\right)^{\delta + \kappa/2} \mathfrak{v}(\tau).$$

Using that $I$ is increasing, we have that

$$|\mathcal{V}(\tau)| \leq \frac{\xi(\tau)}{1 + \delta - \kappa} \left(\frac{\tau}{\tau_0}\right)^{\delta + \kappa/2} I(\tau; \tau_0) + \frac{\xi(\tau)^2}{1 + \delta - \kappa} \left(\frac{\tau}{\tau_0}\right)^{\delta + \kappa/2} \mathfrak{v}(\tau).$$

This proves the second part of the lemma. This leads to the inequality

$$\mathfrak{v}(\tau) \leq \frac{\xi(\tau) I(\tau; \tau_0)}{1 + \delta - \kappa - \xi(\tau)^2} \leq C(\delta, \kappa) \xi(\tau) I(\tau; \tau_0),$$

(using boundedness of $\xi$) which in turn gives

$$|\mathcal{V}(\tau)| \leq C(\delta, \kappa) \xi(\tau) \left(\frac{\tau}{\tau_0}\right)^{\delta + \kappa/2} I(\tau; \tau_0),$$

and which concludes the proof.

**Completing the proof.** We start with

$$I_2(\tau; \tau_0) = \int_{\tau_0}^{\tau} \left(\frac{u}{\tau_0}\right)^{-\delta - \kappa/2} \xi'(u) e^{-(\tau - u)} \, du = \int_{\tau_0}^{\tau} \left(\frac{u}{\tau_0}\right)^{-\delta - \kappa/2} \sqrt{c} u^{-\kappa/2} \mathfrak{m}^{-1/2 + \kappa/2} e^{-(\tau - u)} \, du.$$

Dropping the exponential, we therefore have

$$I_2(\tau; \tau_0) \leq \frac{\tau_0^{-\delta - \kappa + 1}}{(\delta + \kappa - 1) \tau_0^{-\delta - \kappa/2}} \sqrt{c} \mathfrak{m}^{-1/2 + \kappa/2}$$

$$= (\delta + \kappa - 1)^{-1} \sqrt{c} \mathfrak{m}^{-1/2 + \kappa/2} \tau_0^{-\kappa/2 + 1} = \frac{1 - \kappa/2}{\delta + \kappa - 1} \xi(\tau_0; 0).$$

Moreover, this inequality becomes an asymptotic when $\tau/\tau_0 \to \infty$ but $\tau \to 0$.

Turning to the claims on $\Phi$, in terms of $\mathcal{P}$ (122), we have (with $\tau = \mathfrak{m}(1 + t)$ and $\tau_0 = \mathfrak{m}(1 + s)$)

$$\Phi_{11}(t, s) = \mathcal{P}_{11}^2(\tau; \tau_0) \lesssim 1.$$

Using $\xi(\tau_0) \lesssim \sqrt{cm}(1+s)^{1-\kappa/2}$ and $\mathfrak{s}(s) = c(1+s)^{-\kappa}$

$$\mathcal{P}_{12}^2(\tau;\tau_0)\frac{\gamma_2^2\mathfrak{s}(s)}{\mathfrak{m}} \lesssim \xi(\tau_0)^2\frac{\gamma_2^2\mathfrak{s}(s)}{\mathfrak{m}} \lesssim c\gamma_2^2(1+s)^{2(1-\kappa)}$$

which concludes the proof.

$\square$

**Lemma I.3** (Transition). *For any $\epsilon > 0$, there is an $\mathfrak{m}_0$, an $M_1 > 0$ and $M_2 > 0$ so that if $\mathfrak{m} < \mathfrak{m}_0$ so that*

1. *Uniformly for $\epsilon\mathfrak{m}^\mathfrak{p} < \tau_0 < \tau < \frac{1}{\epsilon}\mathfrak{m}^\mathfrak{p}$, the fundamental matrices*

$$\|\mathcal{P}(\tau;\tau_0)\| + \|\mathcal{P}^{-1}(\tau;\tau_0)\| \leq M_1.$$

*Moreover, the fundamental matrix is uniformly close to an explicit expression involving Bessel functions.*

2. *For $\tau_0 < \epsilon^2\mathfrak{m}^\mathfrak{p}$,*

$$\begin{pmatrix} \mathcal{P}_{11}(\tau;\mathfrak{m}) \\ \frac{\delta+\kappa-1}{(1-\kappa/2)\xi(\tau_0;0)}\mathcal{P}_{12}(\tau;\mathfrak{m}) \end{pmatrix} = \left(\frac{2}{\pi}\right)^{\frac{1}{2}}\xi^{-\rho}\left(\begin{pmatrix} \cos(\xi+\omega_\nu(0)) \\ \cos(\xi+\omega_\nu(0)) \end{pmatrix} + O(\epsilon) + O(\xi^{-1})\right).$$

*In this regime, $|\xi|$ is bounded above by a constant depending only on $\epsilon$ and not on $\mathfrak{m}$.*

*Proof.* Starting from (116), we have

$$\frac{\mathrm{d}}{\mathrm{d}\tau}\begin{pmatrix} X \\ \mathcal{Y} \end{pmatrix} = \begin{pmatrix} -1 & -\xi'(\tau) \\ \xi'(\tau) & -\frac{\delta+\kappa/2}{\tau} \end{pmatrix}\begin{pmatrix} X \\ \mathcal{Y} \end{pmatrix}.$$

We change time to $\xi$, which leads to

$$\frac{\mathrm{d}}{\mathrm{d}\xi}\begin{pmatrix} X \\ \mathcal{Y} \end{pmatrix} = \begin{pmatrix} -1/\xi'(\tau) & -1 \\ 1 & -\frac{\delta+\kappa/2}{\tau\xi'(\tau)} \end{pmatrix}\begin{pmatrix} X \\ \mathcal{Y} \end{pmatrix}.$$

We have by definition that

$$\xi = \xi(\tau;\tau_0) = \sqrt{c}\frac{\left(\tau^{1-\kappa/2} - \tau_0^{1-\kappa/2}\right)}{1-\kappa/2}\mathfrak{m}^{-1/2+\kappa/2}.$$

Hence solving for this, we have

$$\tau^{1-\kappa/2} = \frac{\xi}{\sqrt{c}}\mathfrak{m}^{1/2-\kappa/2}(1-\kappa/2) + \tau_0^{1-\kappa/2}.$$

We also have that

$$\tau\xi'(\tau) = \sqrt{c}\tau^{1-\kappa/2}\mathfrak{m}^{-1/2+\kappa/2} = \xi(1-\kappa/2) + \tau_0^{1-\kappa/2}\mathfrak{m}^{-1/2+\kappa/2} \stackrel{\mathrm{def}}{=} (\xi+\xi_0)(1-\kappa/2).$$

We note that $1/\xi'(\tau) = O(\mathfrak{m}^\mathfrak{p})$ and hence, in conclusion,

$$\frac{\mathrm{d}}{\mathrm{d}\xi}\begin{pmatrix} X \\ \mathcal{Y} \end{pmatrix} = \begin{pmatrix} O(\mathfrak{m}^\mathfrak{p}) & -1 \\ 1 & -\frac{a}{\xi+\xi_0} \end{pmatrix}\begin{pmatrix} X \\ \mathcal{Y} \end{pmatrix} \quad \text{where} \quad a = \frac{\delta+\kappa/2}{1-\kappa/2} > 1.$$

We will use continuity of the fundamental solution in the limit as $\mathfrak{m} \to 0$ to solve the equation (note that $|\xi|$ remains bounded independent of $\mathfrak{m}$). Therefore, it suffices to solve the equation where we have taken this error term to 0. As a second order differential equation in $\xi$, this is

$$X''(\xi) + \frac{a}{\xi+\xi_0}X'(\xi) + X(\xi) = 0.$$

From [37, (10.13.4)], with $\nu = (a-1)/2 > 0$ we have the solutions

$$X(\xi) = c_1(\xi+\xi_0)^{-\nu}J_\nu(\xi+\xi_0) + c_2(\xi+\xi_0)^{-\nu}Y_\nu(\xi+\xi_0)$$

where $J_\nu$ and $Y_\nu$ are Bessel functions of the first and second kind, respectively. This leads to the claimed estimates on the fundamental matrix $\mathcal{P}(\tau;\tau_0)$ and an explicit expression in terms of Bessel functions.

**Matching solutions from the entrance regime.** From the initial conditions which come from Lemma I.2, in the case that we start with $(X(0), \mathcal{Y}(0)) = (1, 0)$ we have with $s$ corresponding to $\epsilon \mathfrak{m}^{\mathfrak{p}}$ that $X(s) = 1 + O(\epsilon)$ and $\mathcal{Y}(s) = O(\epsilon)$. Now on sending $\xi_0 \to 0$ we conclude that $c_1 = 1 + O(\epsilon)$ and $c_2 = O(\xi_0^{2\nu})$ (using [37, (10.7.3)]).

If we instead start with $(X(0), \mathcal{Y}(0)) = (0, 1)$ we now get at $s$ (provided $\tau_0/(\epsilon \mathfrak{m}^{\mathfrak{p}}) < \epsilon$)

$$X(s) = \frac{1 - \kappa/2}{\delta + \kappa - 1} \xi(\tau_0; 0)(1 + O(\epsilon))$$

(taking the asymptotic of $I_2$) whereas

$$\mathcal{Y}(s) = O(\xi^2(\tau_0; 0)),$$

which therefore leads to the same $c_1$ and $c_2$ as in the first case, up to rescaling the solution by $\frac{1 - \kappa/2}{\delta + \kappa - 1} \xi(\tau_0; 0)$.

For $\mathcal{Y}$, we have

$$\frac{\mathrm{d}\mathcal{Y}}{\mathrm{d}\xi} + \frac{a}{\xi + \xi_0} \mathcal{Y} = X.$$

Using the integrating factor $\mu(\xi) = (\xi + \xi_0)^a$, we multiply both sides:

$$(\xi + \xi_0)^a \frac{\mathrm{d}\mathcal{Y}}{\mathrm{d}\xi} + a(\xi + \xi_0)^{a-1} \mathcal{Y} = (\xi + \xi_0)^a X.$$

The left side is the derivative of $(\xi + \xi_0)^a \mathcal{Y}$, so

$$\frac{\mathrm{d}}{\mathrm{d}\xi}((\xi + \xi_0)^a \mathcal{Y}) = (\xi + \xi_0)^a X.$$

Therefore

$$\mathcal{Y}(\xi) = (\xi + \xi_0)^{-a} \left( \xi_0^a \mathcal{Y}(0) + \int_0^\xi (s + \xi_0)^a X(s) \, \mathrm{d}s \right).$$

Using [37, (10.22.1)], we have

$$\int_0^\xi (s + \xi_0)^a X(s) \, \mathrm{d}s = c_1(\xi + \xi_0)^{\nu+1} J_{\nu+1}(\xi + \xi_0) + c_2(\xi + \xi_0)^{\nu+1} Y_{\nu+1}(\xi + \xi_0) + O(1),$$

and hence

$$\mathcal{Y}(\xi) = c_1(\xi + \xi_0)^{-\nu} J_{\nu+1}(\xi + \xi_0) + c_2(\xi + \xi_0)^{-\nu} Y_{\nu+1}(\xi + \xi_0) + O((\xi + \xi_0)^{-a}).$$

We recall the following large $\xi$ asymptotic of the Bessel function:

From [37, (10.17.2)], we have

$$\omega(z) = z - \frac{1}{2}\nu\pi - \frac{1}{4}\pi,$$

and from [37, (10.17.3)], as $z \to \infty$ with $\nu$ fixed,

$$J_\nu(z) \sim \left( \frac{2}{\pi z} \right)^{\frac{1}{2}} \cos \omega_\nu + O(z^{-3/2}),$$

$$Y_\nu(z) \sim \left( \frac{2}{\pi z} \right)^{\frac{1}{2}} \sin \omega_\nu + O(z^{-3/2}).$$

Hence we have (for large $\xi$) and with $\omega_\nu = \omega(\xi + \xi_0)$

$$X(\xi) = \left( \frac{2}{\pi} \right)^{\frac{1}{2}} (\xi + \xi_0)^{-\nu - 1/2} \left( (1 + O(\epsilon)) \cos \omega_\nu + O(\epsilon) \sin \omega_\nu + O(\xi^{-1}) \right),$$

$$\mathcal{Y}(\xi) = \left( \frac{2}{\pi} \right)^{\frac{1}{2}} (\xi + \xi_0)^{-\nu - 1/2} \left( (1 + O(\epsilon)) \cos \omega_{\nu+1} + O(\epsilon) \sin \omega_{\nu+1} + O(\xi^{-1}) \right).$$

Note that $\sin(\omega_{\nu+1}) = -\cos(\omega_\nu)$ and $\cos(\omega_{\nu+1}) = \sin(\omega_\nu)$.

Finally, we note that

$$\nu + \frac{1}{2} = \frac{a}{2} = \frac{\delta/2 + \kappa/4}{(1 - \kappa/2)}.$$

We also note that $\xi_0 = \mathfrak{m}^{1/2}$ which tends to $0$.

$\square$

**Lemma I.4** (Bulk). *For any $\epsilon, \mathfrak{c} > 0$, there is an $\mathfrak{m}_0$, if $\mathfrak{m} < \mathfrak{m}_0$ so that*

$$\|\mathcal{P}(\tau; \tau_0) - e^{-(\tau - \tau_0)/2}(\tau/\tau_0)^{-\delta/2-\kappa/4}\mathcal{R}(\tau; \tau_0)\| \le \mathfrak{c}e^{-(\tau - \tau_0)/2}(\tau/\tau_0)^{-\delta/2-\kappa/4}.$$

*Proof.* We recall that the range of $\tau$ is given by

$$\frac{1}{\epsilon}\mathfrak{m}^{\mathfrak{p}} \le \tau < \frac{1}{\epsilon}.$$

We first estimate $\mathcal{U}(\tau; \tau_0)$, recalling that

$$\frac{\mathrm{d}}{\mathrm{d}\tau}\mathcal{U}(\tau; \tau_0) = \mathcal{R}^{-1}\begin{pmatrix} -1 & 0 \\ 0 & -\frac{\delta+\kappa/2}{\tau} \end{pmatrix}\mathcal{R}\mathcal{U}(\tau; \tau_0).$$

Using trig identities, there are matrices with non-constant trig polynomials $\mathcal{W}_i(\xi)$ (having no constant terms) so that

$$\mathcal{R}^{-1}\begin{pmatrix} -1 & 0 \\ 0 & -\frac{\delta+\kappa/2}{\tau} \end{pmatrix}\mathcal{R} = \left(\left(\frac{-1}{2} - \frac{\delta/2 + \kappa/4}{\tau}\right)\mathrm{Id} + \mathcal{W}_1(\xi) + \frac{1}{\tau}\mathcal{W}_2(\xi)\right). \tag{124}$$

We need an integration-by-parts estimate for oscillatory integrals.

**Lemma I.5** (Integration-by-parts estimate). *Let $p$ be a positive integer. There is a constant $C(\kappa)$ so that for any absolutely continuous $g$,*

$$\left|\int_{\tau_0}^{\tau} e^{ip\xi(u)}g(u)\,\mathrm{d}u\right| \le \frac{C(\kappa)}{\xi'(\tau_0)}\left(\int_{\tau_0}^{\tau}|g'(u)|\,\mathrm{d}u + \sup_{u \in [\tau_0, \tau]}|g(u)|\right).$$

*Proof.* We recall that $\xi'(\tau) = \tau^{-\kappa/2}\mathfrak{m}^{-1/2+\kappa/2}$. Applying integration by parts, we have

$$\int_{\tau_0}^{\tau} e^{ip\xi(u)}g(u)\,\mathrm{d}u = \int_{\tau_0}^{\tau} e^{ip\xi(u)}\frac{ip\xi'(u)g(u)}{ip\xi'(u)}\,\mathrm{d}u$$

$$= \frac{e^{ip\xi(u)}g(u)}{ip\xi'(u)}\Big|_{\tau_0}^{\tau} - \int_{\tau_0}^{\tau} e^{ip\xi(u)}\frac{\mathrm{d}}{\mathrm{d}u}\left(\frac{g(u)}{ip\xi'(u)}\right)\,\mathrm{d}u.$$

Expanding the derivatives and bounding, brings us to the claim. $\square$

We now divide the range of time into two parts.

**Large $\tau$:** $\tau > \tau_0 > \epsilon$. Suppose that $\tau > \tau_0 > \epsilon$ as well. By standard approximation arguments, it follows that for any bounded continuous function $f$ on $[-1/\epsilon, 1/\epsilon]$,

$$\left|\int_{-1/\epsilon}^{1/\epsilon} e^{ip\xi(u)}f(u)\,\mathrm{d}u\right| \to 0 \quad \text{and} \quad \left|\int_{-1/\epsilon}^{1/\epsilon} e^{ip\xi(u)}u^{-1}f(u)\,\mathrm{d}u\right| \to 0$$

as $\mathfrak{m} \to 0$. It follows from (124) that the matrix converges weak-$*$ as $\mathfrak{m} \to 0$.

$$\mathcal{R}^{-1}(\tau; 0)\begin{pmatrix} -1 & 0 \\ 0 & -\frac{\delta+\kappa/2}{\tau} \end{pmatrix}\mathcal{R}(\tau; 0) \xrightarrow{\mathfrak{m} \to 0} \left(\frac{-1}{2} - \frac{\delta/2 + \kappa/4}{\tau}\right)\mathrm{Id}.$$

Solving differential equations is continuous with respect to weak-$*$ convergence, so it follows that

$$\mathcal{U}(\tau; 0) \xrightarrow{\mathfrak{m} \to 0} e^{-\tau/2}\tau^{-\delta/2-\kappa/4}\mathrm{Id}.$$

And therefore for any $\mathfrak{c} > 0$ (including those which depend on $\epsilon$) there is $\mathfrak{m}$ sufficiently large so that

$$\|\mathcal{U}(\tau; 0)e^{\tau/2}\tau^{\delta/2+\kappa/4} - \mathrm{Id}\| \le \mathfrak{c} \quad \text{and} \quad \|\mathcal{U}(\tau; 0) - e^{-\tau/2}\tau^{-\delta/2-\kappa/4}\mathrm{Id}\| \le \mathfrak{c}.$$

**Small $\tau$:** $\frac{1}{\epsilon}\mathfrak{m}^{\mathfrak{p}} < \tau < \epsilon$. We choose $\frac{1}{\epsilon}\mathfrak{m}^{\mathfrak{p}} \leq \tau_0 < \tau < \epsilon$. Using (124), we change variables to let

$$\mathcal{V}(\tau;\tau_0) = \mathcal{U}(\tau;\tau_0)e^{(\tau-\tau_0)/2}(\tau/\tau_0)^{\delta/2+\kappa/4}.$$

Then we have

$$\frac{\mathrm{d}}{\mathrm{d}\tau}\mathcal{V} = \left(\mathcal{W}_1(\xi) + \frac{1}{\tau}\mathcal{W}_2(\xi)\right)(\mathcal{V}).$$

Hence if we let $\|\cdot\|_{\max}$ be the maximum entry norm, and we apply Lemma I.5 entrywise, we get

$$
\begin{aligned}
\int_{\tau_0}^\tau \left\|\frac{\mathrm{d}}{\mathrm{d}u}\mathcal{V}(u;\tau_0)\right\|_{\max} &\leq \frac{C(\kappa)}{\xi'(\tau_0)}\left(\int_{\tau_0}^\tau \left\|\frac{\mathrm{d}}{\mathrm{d}u}\mathcal{V}(u;\tau_0)\right\|_{\max}\mathrm{d}u + \sup_{u\in[\tau_0,\tau]}\|\mathcal{V}(u;\tau_0)\|_{\max}\right) \\
&+ \frac{C(\kappa)}{\xi'(\tau_0)}\left(\int_{\tau_0}^\tau \left\|\frac{\mathrm{d}}{\mathrm{d}u}\frac{\mathcal{V}(u;\tau_0)}{u}\right\|_{\max}\mathrm{d}u + \sup_{u\in[\tau_0,\tau]}\left\|\frac{\mathcal{V}(u;\tau_0)}{u}\right\|_{\max}\right).
\end{aligned}
\tag{125}
$$

Now we use that

$$\frac{\mathrm{d}}{\mathrm{d}u}\left(\frac{\mathcal{V}(u;\tau_0)}{u}\right) = \frac{1}{u}\frac{\mathrm{d}}{\mathrm{d}u}(\mathcal{V}(u;\tau_0)) - \frac{1}{u^2}\mathcal{V}(u;\tau_0).$$

And hence

$$\left\|\frac{\mathrm{d}}{\mathrm{d}u}\left(\frac{\mathcal{V}(u;\tau_0)}{u}\right)\right\|_{\max} \leq \frac{1}{\tau_0}\left\|\frac{\mathrm{d}}{\mathrm{d}u}\mathcal{V}(u;\tau_0)\right\|_{\max} + \frac{1}{u^2}\sup_{u\in[\tau_0,\tau]}\|\mathcal{V}(u;\tau_0)\|_{\max}.$$

We also use that

$$\sup_{u\in[\tau_0,\tau]}\|\mathcal{V}(u;\tau_0)\|_{\max} \leq 1 + \int_{\tau_0}^\tau \left\|\frac{\mathrm{d}}{\mathrm{d}u}\mathcal{V}(u;\tau_0)\right\|_{\max}\mathrm{d}u.$$

Combining all of these estimates, and rearranging (125), we have

$$\left(\int_{\tau_0}^\tau \left\|\frac{\mathrm{d}}{\mathrm{d}u}\mathcal{V}(u;\tau_0)\right\|_{\max}\mathrm{d}u\right)\left(1 - \frac{2C(\kappa)}{\xi'(\tau_0)} - \frac{2C(\kappa)}{\xi'(\tau_0)\tau_0}\right) \leq \frac{C(\kappa)}{\xi'(\tau_0)}.$$

Recall that

$$\xi'(\tau_0) = \tau_0^{-\kappa/2}\mathfrak{m}^{-1/2+\kappa/2},$$

and hence using that $\frac{1}{\epsilon}\mathfrak{m}^{\mathfrak{p}} \leq \tau_0$

$$\xi'(\tau_0)\tau_0 = \tau_0^{1-\kappa/2}\mathfrak{m}^{-1/2+\kappa/2} \geq \epsilon^{-1+\kappa/2}\mathfrak{m}^{(1-\kappa)/2\mathfrak{p}}\mathfrak{m}^{-1/2+\kappa/2} \geq \epsilon^{-1+\kappa/2}.$$

We conclude that for $\epsilon$ sufficiently small,

$$\left(\int_{\tau_0}^\tau \left\|\frac{\mathrm{d}}{\mathrm{d}u}\mathcal{V}(u;\tau_0)\right\|_{\max}\mathrm{d}u\right) \leq 2C(\kappa)\tau_0^{\kappa/2}\mathfrak{m}^{1/2-\kappa/2}.$$

This leads to the estimate

$$\|\mathcal{V}(\tau;\tau_0) - \mathrm{Id}\|_{\max} \leq 2C(\kappa)\tau_0^{\kappa/2}\mathfrak{m}^{1/2-\kappa/2},$$

and hence for $\mathcal{U}$ we have

$$\|\mathcal{U}(\tau;\tau_0) - e^{-(\tau-\tau_0)/2}(\tau/\tau_0)^{-\delta/2-\kappa/4}\mathrm{Id}\| \leq 2C(\kappa)\tau_0^{\kappa/2}\mathfrak{m}^{1/2-\kappa/2}e^{-(\tau-\tau_0)/2}(\tau/\tau_0)^{-\delta/2-\kappa/4}.$$

**Combining the estimates.** Using that $\mathcal{P}$ and $\mathcal{U}$ only differ by a rotation, we conclude that for any $\mathfrak{c}$, for any $\tau_0 < \tau$ in the regime.

$$\|\mathcal{P}(\tau;\tau_0) - e^{-(\tau-\tau_0)/2}(\tau/\tau_0)^{-\delta/2-\kappa/4}\mathcal{R}(\tau;\tau_0)\| \leq \mathfrak{c}e^{-(\tau-\tau_0)/2}(\tau/\tau_0)^{-\delta/2-\kappa/4}.$$

$\square$

Finally, for larger $\tau$, while sharp asymptotics are possible, it suffices to bound the decay of the solutions. We do this with two separate estimates.

**Lemma I.6** (Exponential Decay). *Suppose $\delta > 2$. There is a constant $\mathfrak{c} > 0$ and $\epsilon_0$ so that for all $\epsilon < \epsilon_0$ and $\tau > \tau_0 > \frac{1}{\epsilon}$, and $\xi'(\tau) = \sqrt{c}\tau^{-\kappa/2}\mathfrak{m}^{-1/2+\kappa/2} \geq 1/\epsilon$*

$$\|\mathcal{P}(\tau;\tau_0)\| \leq (1 + \mathfrak{c}\epsilon)e^{-(1/2 - \mathfrak{c}\epsilon)(\tau - \tau_0)}(\tau/\tau_0)^{-\delta/2 - \kappa/4}.$$

*Proof.* We compute the eigenvalues of the matrix that appears in (117). We have

$$\lambda_\pm = \frac{-1 - \frac{\delta + \kappa/2}{\tau} \pm \sqrt{(1 + \frac{\delta + \kappa/2}{\tau})^2 - 4(\xi')^2}}{2}.$$

In particular in the regime in which we operate, we have the eigenvalues are complex conjugate pairs (as $\xi'$ is large), and in fact we have $|\lambda_\pm - \pm i\xi'| \leq 1$. We introduce a change of basis matrix $H$ so that

$$\frac{\mathrm{d}}{\mathrm{d}\tau}\mathcal{P} = H^{-1}\begin{pmatrix} \lambda_+ & 0 \\ 0 & \lambda_- \end{pmatrix}H\mathcal{P}.$$

We note that this eigenvector matrix is within $\epsilon$ of $\begin{pmatrix} 1 & i \\ i & 1 \end{pmatrix}$, owing to the magnitude of $\xi'$. In particular we have $H^*H = \mathrm{Id} + O(\epsilon)$ and furthermore $\|H'\| = O(\epsilon)$. Hence differentiating

$$\frac{\mathrm{d}}{\mathrm{d}\tau}(\mathcal{P}^*H^*H\mathcal{P}) = (\mathcal{P}^*H^*(\Lambda + \Lambda^*)H\mathcal{P}) + \left(\mathcal{P}^*\left(\frac{\mathrm{d}}{\mathrm{d}\tau}(H^*H)\right)\mathcal{P}\right).$$

Thus for any fixed vector $v \in \mathbb{C}^2$ if $w(\tau) = \|H\mathcal{P}v\|^2$

$$\frac{\mathrm{d}}{\mathrm{d}\tau}w \leq \left(-1 - \frac{\delta + \kappa/2}{\tau} + O(\epsilon)\right)w,$$

and hence

$$\|\mathcal{P}v\|^2 \leq (1 + O(\epsilon))e^{-(1-\epsilon)(\tau - \tau_0)}(\tau/\tau_0)^{-\delta - \kappa/2}.$$

$\square$

**Lemma I.7** (Slow Decay). *Suppose $\delta > 2$. There is a constant $c > 0$ and $\epsilon_0$ so that for all $\epsilon < \epsilon_0$ and $\tau > \tau_0 > \frac{1}{\epsilon}$,*

$$\|\mathcal{P}(\tau;\tau_0)\| \leq (\tau/\tau_0)^{-\delta/2 - \kappa/4}.$$

*We further have an improved estimate when $\xi'(\tau_0) \leq \frac{1}{4}$ is small*

$$\max\{|\mathcal{P}_{1,1}(\tau;\tau_0)|, |\mathcal{P}_{1,2}(\tau;\tau_0)|\} \leq e^{-(\tau - \tau_0)} + C(\kappa, \delta)(\xi'(\tau))^2\exp\left(\int_{\tau_0}^\tau -(\xi'(u))^2\,\mathrm{d}u\right).$$

*Proof.* From the equation (120) we have

$$\frac{\mathrm{d}}{\mathrm{d}\tau}\left(U^2(\tau;\tau_0) + V^2(\tau;\tau_0)\right) = \begin{pmatrix} U & V \end{pmatrix}\mathcal{R}^\top\begin{pmatrix} -1 & 0 \\ 0 & -\frac{\delta + \kappa/2}{\tau} \end{pmatrix}\mathcal{R}\begin{pmatrix} U \\ V \end{pmatrix}.$$

By assumption we have $\tau > \frac{1}{\epsilon}$, and hence the matrix in the middle is dominated by $-\frac{\delta + \kappa/2}{\tau}I$. Therefore

$$\frac{\mathrm{d}}{\mathrm{d}\tau}\left(U^2(\tau;\tau_0) + V^2(\tau;\tau_0)\right) \leq -\frac{\delta + \kappa/2}{\tau}\left(U^2(\tau;\tau_0) + V^2(\tau;\tau_0)\right).$$

Integrating this from $\tau_0$ to $\tau$ we get for any unit vector $v \in \mathbb{R}^2$

$$\|\mathcal{U}(\tau;\tau_0)v\|^2 \leq (\tau/\tau_0)^{-\delta - \kappa/2},$$

and hence the operator norm of $\mathcal{U}$ decays the same way. Since $\mathcal{P}$ is a rotation of $\mathcal{U}$, it decays at the same rate.

**Improved estimate.** We start with (117)

$$\frac{\mathrm{d}}{\mathrm{d}\tau}\begin{pmatrix} X \\ \mathcal{Y} \end{pmatrix} = \begin{pmatrix} -1 & -\xi' \\ -\xi' & -\frac{\delta+\kappa/2}{\tau} \end{pmatrix}\begin{pmatrix} X \\ \mathcal{Y} \end{pmatrix}.$$

We set

$$\lambda_\pm = \frac{-1 \pm \sqrt{1 - 4(\xi')^2}}{2},$$

where we note the radical is real for the regime chosen and it approaches 1 as $\xi' \to 0$. Define $W_\pm$ by

$$W_\pm = X + \sqrt{\lambda_\mp/\lambda_\pm}\mathcal{Y}.$$

From a direct computation

$$\frac{\mathrm{d}}{\mathrm{d}\tau}\log(\lambda_-/\lambda_+) = \frac{\kappa}{\tau}\frac{1}{\sqrt{1 - 4(\xi')^2}}.$$

Hence we get

$$\frac{\mathrm{d}}{\mathrm{d}\tau}W_\pm = \lambda_\pm W_\pm + (W_+ - W_-)O(1/\tau),$$

where $O(1/\tau)$ is bounded by $C(\kappa, \delta)/\tau$. Uniformly over the range of $\tau$ considered, we can bound $\frac{1}{u} \le \epsilon\xi'(u)$, and so

$$\frac{\mathrm{d}}{\mathrm{d}\tau}\left(W_+^2 + W_-^2\right) \le 2\left(\lambda_+ + \frac{C(\kappa,\delta)}{\tau}\right)\left(W_+^2 + W_-^2\right) \le 2\left(-(\xi'(\tau))^2\right)\left(W_+^2 + W_-^2\right)$$

Integrating from a $\tau_0$ with $\xi'$ at $\tau_0$ at most $\frac{1}{4}$,

$$\left(W_+^2 + W_-^2\right)(\tau) \le \left(W_+^2 + W_-^2\right)(\tau_0)\exp\left(2\int_{\tau_0}^\tau -(\xi'(u))^2\,\mathrm{d}u\right).$$

We have

$$\mathcal{Y}(\tau) = \sqrt{\lambda_+\lambda_-}\frac{W_+ - W_-}{\lambda_+ - \lambda_-},$$

so that for an absolute constant $C > 0$

$$|\mathcal{Y}(\tau)| \le C\xi'(\tau)\sqrt{\left(W_+^2 + W_-^2\right)(\tau)}$$
$$\le C\xi'(\tau)\sqrt{\left(W_+^2 + W_-^2\right)(\tau_0)}\exp\left(\int_{\tau_0}^\tau -(\xi'(u))^2\,\mathrm{d}u\right).$$

Returning to the differential equation,

$$\frac{\mathrm{d}}{\mathrm{d}\tau}\left(e^\tau X(\tau)\right) = -\xi'(\tau)e^\tau\mathcal{Y}(\tau).$$

Integrating both sides and bounding, we arrive at

$$|X(\tau)| \le |X(\tau_0)|e^{-(\tau-\tau_0)}$$
$$+ C\left(\xi'(\tau)\right)^2\sqrt{\left(W_+^2 + W_-^2\right)(\tau_0)}\exp\left(\int_{\tau_0}^\tau -(\xi'(u))^2\,\mathrm{d}u\right).$$

$\square$

## I.2 Computing the forcing function

In this section, we prove the scaling law for DANA-decaying as parametrized in Parametrization I.1 using the estimates in Theorem I.1.

We begin by estimating the forcing function.

**Proposition I.1** (Forcing function). *Let $\alpha > \frac{1}{2}$, $2\alpha + 2\beta > 1$, $\alpha, \beta \neq \frac{1}{2}$, $\alpha + 1 > \beta$. Suppose that $2\rho \stackrel{def}{=} \frac{\delta + \frac{1}{4\alpha}}{1 - \frac{1}{4\alpha}} > \frac{2\alpha + 2\beta - 1}{\alpha}$. Moreover, denote $\vartheta(t) \stackrel{def}{=} 1 + 2\gamma_2 Bt + \left(\int_0^t \sqrt{\gamma_3(s)B}\,ds\right)^2 \asymp 1 + \left(\int_0^t \sqrt{\gamma_3(s)B}\,ds\right)^2$ and suppose Parametrization I.1 holds. Then there exists some $C(\alpha, \beta) > 0$ such that for any $t \geq 0$ and $d$ large enough*

$$\frac{1}{C}\left(\mathcal{F}_0(t) + \mathcal{F}_{pp}(t) + \mathcal{F}_{ac}(t)\right) \leq \mathcal{F}(t) \leq C\left(\mathcal{F}_0(t) + \mathcal{F}_{pp}(t) + \mathcal{F}_{ac}(t)\right)$$

*where*

$$\mathcal{F}_0 \asymp d^{-2\alpha + (1-2\beta)_+}, \quad \mathcal{F}_{pp}(t) \asymp \vartheta(t)^{-1 - \frac{2\beta - 1}{2\alpha}},$$

$$\text{and} \quad \mathcal{F}_{ac}(t) \begin{cases} \asymp \vartheta(t)^{-1 + \frac{1}{2\alpha}}, & \text{if } 2\alpha > 1, \, 2\beta > 1 \\ \lesssim \mathcal{F}_0, & \text{if } 2\alpha < 1, \, 2\beta > 1 \\ = 0, & \text{else.} \end{cases}$$

*Proof.* We define $\mathcal{F}_0, \mathcal{F}_{pp}, \mathcal{F}_{ac}$ as in Section D and use the estimate on $\Phi_{11}^{\sigma^2}(t, 0)$ from Theorem I.1 and more precisely Equation (110). $\mathcal{F}_0$ is unchanged, however for example for $\mathcal{F}_{pp}(t)$ we compute

$$\mathcal{F}_{pp}(t) \stackrel{def}{=} \int_0^1 \sigma^{1 + \frac{2\beta - 1}{\alpha}} \left( e^{-\gamma_2 B\sigma^2(1+t)}(1 + \frac{\sqrt{\gamma_2 cB\sigma^2}}{1 - \frac{\kappa}{2}}((1+t)^{1 - \frac{\kappa}{2}} - 1))^{-2\rho} \right.$$

$$\left. + \mathcal{O}(\epsilon(\gamma_2 B\sigma^2)^{\delta + \frac{\kappa}{2}}((\gamma_2 B\sigma^2(1+t)) \wedge 1)^{-\delta - \frac{\kappa_3}{2}}) \right)$$

$$\asymp \int_{\sigma=0}^{\min\{\frac{1}{\sqrt{\gamma_2 Bt}}, \frac{1}{\sqrt{\gamma_2 Bc(1+t)^{1 - \frac{\kappa}{2}}}}, 1\}} \sigma^{1 + \frac{2\beta - 1}{\alpha}}\,d\sigma$$

$$\asymp \vartheta(t)^{-1 - \frac{2\beta - 1}{2\alpha}},$$

where we used that $-2\rho + 1 - \frac{2\beta - 1}{\alpha} < -1 \iff 2\rho > \frac{2\alpha + 2\beta - 1}{\alpha}$. A similar computation on $\mathcal{F}_{ac}$ gives the result, with the additional condition $2\rho > 2 - \frac{1}{\alpha}$ which is automatically satisfied since $\frac{2\alpha + 2\beta - 1}{\alpha} > 2 - \frac{1}{\alpha}$.

Finally, to bound $\mathcal{F}$ using $\mathcal{F}_0(t) + \mathcal{F}_{pp}(t) + \mathcal{F}_{ac}(t)$ we proceed with a similar proof as Proposition H.16 by noting that in the range of interest (integral on second line), $\Phi_{11}^{\sigma^2}(t, 0)$ satisfies the hypothesis in Proposition D.13 (it is constant).

$\square$

## I.3  Stability condition for DANA-decaying.

We can now prove a stability condition for DANA-decaying.

**Proposition I.2** (Stability of DANA-decaying using Parametrization I.1 under Assumption 6). *Consider Parametrization I.1 under Assumption 6. Suppose that*

$$2\rho \stackrel{def}{=} \frac{\delta + \frac{1}{4\alpha}}{1 - \frac{1}{4\alpha}} > \max\left\{\frac{2\alpha + 2\beta - 1}{\alpha}, 4 - \frac{1}{\alpha}\right\} \quad \text{and} \quad \delta + \frac{\kappa}{2} > 1. \tag{126}$$

*Then we have for any $\epsilon > 0$ that there exists $\mathfrak{g}(\kappa, \epsilon) > 0$ and $d_0$ large enough, such that for any $d \geq d_0$ if $\gamma_2 = \mathfrak{g}$ and $c \leq \mathfrak{g}$, then $\sup_{t \geq 0} \int_0^t \mathcal{K}(t, s)\,ds < \epsilon$. In particular, since $\mathcal{F}(t)$ is bounded (indep of $d$), we have $\sup_{d \geq d_0} \|\mathcal{P}\|_\infty < \infty$.*

*Proof.* We again consider the estimates in Theorem I.1. The goal is to have sufficient conditions so that $\int_0^t \mathcal{K}(t, s)\,ds < \epsilon$ for any $t \geq 0$.

We compute the first term by bounding the oscillatory $\omega^1(\xi, \xi_0)$ by constants and $\frac{1+\xi}{1+\xi_0} \leq 1$ to get that $\Phi_{11}(t, s) \lesssim e^{-(\tau(t) - \tau(s))}$. For $M > 0$ large,

$$\int_0^t \mathcal{K}^1(t, s) \, \mathrm{d}s \lesssim \gamma_2^2 B \int_0^t \int_{\sigma = \frac{1}{M} d^{-\alpha}}^1 \sigma^{3 - \frac{1}{\alpha}} e^{-\gamma_2 B \sigma^2 (t-s)} \, \mathrm{d}\sigma \, \mathrm{d}s$$

$$+ \gamma_2^2 B \int_{s=0}^t \int_{\sigma = \frac{\sqrt{\gamma_3(t)B}}{\gamma_2 B}}^1 \sigma^{3 - \frac{1}{\alpha}} \left( \frac{\sqrt{\gamma_3(t)B}}{\gamma_2 B \sigma} \right)^4 e^{-\gamma_3(t)/\gamma_2 (t-s)} \, \mathrm{d}\sigma \, \mathrm{d}s$$

$$\lesssim \gamma_2 \int_{\sigma = \frac{1}{M} d^{-\alpha}}^1 \sigma^{1 - \frac{1}{\alpha}} [1 - e^{-\gamma_2 B \sigma^2 t}] \, \mathrm{d}\sigma + \gamma_2 \int_{\sigma = \frac{\sqrt{\gamma_3(t)B}}{\gamma_2 B}}^1 \gamma_2 \sigma^{1 - 1/\alpha} \, \mathrm{d}\sigma$$

$$\lesssim \gamma_2.$$

Here we used that $\int_{s=0}^t e^{-\gamma_3(t)/\gamma_2 (t-s)} \, \mathrm{d}s \lesssim \frac{\gamma_2}{\gamma_3(t)}$ and that $\left( \frac{\sqrt{\gamma_3(t)B}}{\gamma_2 B \sigma} \right)^4 \lesssim \left( \frac{\sqrt{\gamma_3(t)B}}{\gamma_2 B \sigma} \right)^2 = \frac{\gamma_3(t)}{\gamma_2} \frac{1}{\gamma_2 B \sigma^2}$.

For the second term we do the same to obtain that $\Phi_{12}(t, s) \lesssim e^{-(\tau(t) - \tau(s))} \frac{\gamma_3(s)}{B\sigma^2}$.

We write for $\gamma_2 B(1 + t) \lesssim d^{2\alpha}$

$$\int_0^t \mathcal{K}^2(t, s) \, \mathrm{d}s \lesssim B \int_0^t \int_{\sigma = \frac{1}{M} d^{-\alpha}}^1 \sigma^{3 - \frac{1}{\alpha}} \frac{\gamma_3}{B\sigma^2} (1 + s)^{-\kappa} e^{-\gamma_2 B \sigma^2 (t-s)} \, \mathrm{d}\sigma \, \mathrm{d}s$$

$$+ B \int_0^t \int_{\sigma = \frac{\sqrt{\gamma_3(t)B}}{\gamma_2 B}}^1 \sigma^{3 - \frac{1}{\alpha}} \left( \frac{\sqrt{\gamma_3(t)B}}{\gamma_2 B \sigma} \right)^4 \frac{\gamma_3}{B\sigma^2} (1 + s)^{-\kappa} e^{-\gamma_3(t)(t-s)/\gamma_2} \, \mathrm{d}\sigma \, \mathrm{d}s$$

$$\lesssim \gamma_3 \int_{\sigma = \frac{1}{M} d^{-\alpha}}^1 \sigma^{1 - \frac{1}{\alpha}} \min\{ (1 + t)^{-\kappa} \frac{1}{\gamma_2 B \sigma^2}, (1 + t)^{-\kappa + 1} \} \, \mathrm{d}\sigma$$

$$\lesssim \gamma_3 \int_{\sigma = \frac{1}{M} d^{-\alpha}}^{\frac{1}{\sqrt{\gamma_2 B(1+t)}}} \sigma^{1 - \frac{1}{\alpha}} (1 + t)^{-\kappa + 1} \, \mathrm{d}\sigma + \gamma_3 \int_{\frac{1}{\sqrt{\gamma_2 B(1+t)}}}^1 (1 + t)^{-\kappa} \frac{\sigma^{-1 - \frac{1}{\alpha}}}{\gamma_2 B} \, \mathrm{d}\sigma$$

$$\lesssim \gamma_3 (1 + t)^{-\kappa + 1} (\gamma_2 B(1 + t))^{-1 + \frac{1}{2\alpha}} + \frac{\gamma_3}{\gamma_2 B} (1 + t)^{-\kappa} (\gamma_2 B(1 + t))^{\frac{1}{2\alpha}}.$$

Here we bounded for $\sigma \geq \frac{\sqrt{\gamma_3(t)B}}{\gamma_2 B}$,

$$\int_0^t \sigma^{3 - \frac{1}{\alpha}} \left( \frac{\sqrt{\gamma_3(t)B}}{\gamma_2 B \sigma} \right)^4 \frac{\gamma_3}{B\sigma^2} (1 + s)^{-\kappa} e^{-\gamma_3(t)(t-s)/\gamma_2} \, \mathrm{d}s$$

$$\lesssim \min\{ \frac{\gamma_2}{\gamma_3(t)} \left( \frac{\sqrt{\gamma_3(t)B}}{\gamma_2 B \sigma} \right)^4, \gamma_3(t) t \}$$

$$\lesssim \min\{ (1 + t)^{-\kappa} \frac{1}{\gamma_2 B \sigma^2}, (1 + t)^{-\kappa + 1} \}.$$

When $\gamma_2 B(1 + t) \gtrsim d^{2\alpha}$ the first term vanishes and we obtain

$$\int_0^t \mathcal{K}^2(t, s) \, \mathrm{d}s \lesssim \frac{\gamma_3}{\gamma_2 B} (1 + t)^{-\kappa} \int_{\frac{1}{M} d^{-\alpha}}^1 \sigma^{-1 - \frac{1}{\alpha}} \, \mathrm{d}\sigma$$

$$\lesssim \frac{\gamma_3}{\gamma_2 B} (1 + t)^{-\kappa} \times d.$$

Evaluating at the worst case $t \asymp \frac{d^{2\alpha}}{\gamma_2 B}$ yields the stability condition

$$\frac{\gamma_3}{\gamma_2 B} \left( \frac{d^{2\alpha}}{\gamma_2 B} \right)^{-\kappa} \times d \lesssim 1. \tag{127}$$

$\square$

## I.4 Kernel function

Now that we have shown stability of Parametrization I.1 under Assumption 6, we can proceed with computing the kernel function. For that we define upper on the solutions of the ODE:

$$\bar{\Phi}_{11}^{\sigma}(t,s) \stackrel{\text{def}}{=} e^{-(\tau(t)-\tau(s))}\left(\frac{1+\xi(t)}{1+\xi(s)}\right)^{-2\rho}$$

and $\bar{\Phi}_{12}^{\sigma}(t,s) \stackrel{\text{def}}{=} \begin{cases} e^{-(\tau(t)-\tau(s))}\left(\frac{1+\xi(t)}{1+\xi(s)}\right)^{-2\rho}\frac{\gamma_3(s)}{B\sigma^2} + \Omega(t)^2 e^{-\int_{\tau_0}^{\tau}\Omega(u)\,du}\left(\frac{1+\xi(t)}{1+\xi(s)}\right)^{-2\rho} \\ \quad \text{if } \xi(s) > 1 \\ e^{-(\tau(t)-\tau(s))}\left(\frac{1+\xi(t)}{1+\xi(s)}\right)^{-2\rho}\frac{\gamma_3(s)}{B\sigma^2}(\sigma\sqrt{\gamma_3(s)B}(1+s))^2 + \Omega(t)^2 e^{-\int_{\tau_0}^{\tau}\Omega(u)\,du}\left(\frac{1+\xi(t)}{1+\xi(s)}\right)^{-2\rho} \\ \quad \text{if } \xi(s) \le 1. \end{cases}$

We define accordingly for all $t \ge s \ge 0$

$$\bar{\mathcal{K}}^1(t,s) \stackrel{\text{def}}{=} \int_0^1 \gamma_2^2 B \bar{\Phi}_{11}^{\sigma}(t,s)\,d\mu_{\mathcal{K}}(\sigma^2)$$

$$\bar{\mathcal{K}}^2(t,s) \stackrel{\text{def}}{=} \int_0^1 \gamma_1^2 B \bar{\Phi}_{12}^{\sigma}(t,s)\,d\mu_{\mathcal{K}}(\sigma^2)$$

$$\bar{\mathcal{K}}(t,s) \stackrel{\text{def}}{=} \bar{\mathcal{K}}^1(t,s) + \bar{\mathcal{K}}^2(t,s),$$

$$\bar{\mathcal{K}}_{pp}^1(t,s) \stackrel{\text{def}}{=} \int_0^1 \gamma_2^2 B \bar{\Phi}_{11}^{\sigma}(t,s)\,d\mu_{\mathcal{K}_{pp}}(\sigma^2)$$

$$\bar{\mathcal{K}}_{pp}^2(t,s) \stackrel{\text{def}}{=} \int_0^1 \gamma_1^2 B \bar{\Phi}_{12}^{\sigma}(t,s)\,d\mu_{\mathcal{K}_{pp}}(\sigma^2)$$

$$\text{and} \quad \bar{\mathcal{K}}_{pp}(t,s) \stackrel{\text{def}}{=} \bar{\mathcal{K}}_{pp}^1(t,s) + \bar{\mathcal{K}}_{pp}^2(t,s).$$

**Proposition I.3** (Upper-bound on Kernel). *Consider Parametrization I.1 under Assumption 6. Then an upper-bound on the kernel function is*

$$\mathcal{K}(t,s) \lesssim \bar{\bar{\mathcal{K}}}_{pp}^1(t,s) + \bar{\bar{\mathcal{K}}}_{pp}^2(t,s)$$

*where for some $\tilde{\delta}$ with $\tilde{\delta} \stackrel{\delta \to \infty}{\to} \infty$*

$$\bar{\bar{\mathcal{K}}}_{pp}^1(t,s) = \gamma_2^2 B\left((1+\gamma_2 B(t-s))^{-2+\frac{1}{2\alpha}}\left(\frac{1+t}{1+s}\right)^{-\tilde{\delta}} + (\sqrt{\gamma_2 cB}(1+t)^{1-\frac{\kappa}{2}})^{-4+\frac{1}{\alpha}}\right),$$

$$\bar{\bar{\mathcal{K}}}_{pp}^2(t,s) = \gamma_2 c(1+s)^{-\kappa}\Bigg((1+(\gamma_2 B(t-s)))^{-1+\frac{1}{2\alpha}}\left(\frac{1+t}{1+s}\right)^{-\tilde{\delta}} +$$

$$(\sqrt{\gamma_2 cB}(1+t)^{1-\frac{\kappa}{2}})^{-4+\frac{1}{\alpha}}(\sqrt{\gamma_3(s)B}(1+s))^2\Bigg).$$

*Proof.* We know since $\mu_{\mathcal{K}}$ is a positive measure and $\Phi_{11}^{\sigma}(t,s) \lesssim \bar{\Phi}_{11}^{\sigma}(t,s)$, $\Phi_{12}^{\sigma}(t,s) \lesssim \bar{\Phi}_{12}^{\sigma}(t,s)$ that

$\mathcal{K}(t,s) \lesssim \bar{\mathcal{K}}(t,s)$. There remains to upper-bound $\bar{\mathcal{K}}^1(t,s)$, $\bar{\mathcal{K}}^2(t,s)$. For that, notice that since $\bar{\Phi}_{11}^{\sigma^2}(t,s), \bar{\Phi}_{12}^{\sigma^2}(t,s)$ are well-behaved in the $\sigma$ variable, then using Proposition D.14 and by a similar proof as in Proposition H.19, we have $\forall t \ge s \ge 0$

$$\bar{\mathcal{K}}^1(t,s) \lesssim \bar{\mathcal{K}}_{pp}^1(t,s)$$
$$\bar{\mathcal{K}}^2(t,s) \lesssim \bar{\mathcal{K}}_{pp}^2(t,s).$$

We only need to upper-bound $\bar{\mathcal{K}}^1_{pp}(t,s)$, $\bar{\mathcal{K}}^2_{pp}(t,s)$. For the first term we have in the case $\frac{\sqrt{\gamma_3(t)B}}{\gamma_2 B} \gtrsim \frac{1}{\sqrt{\gamma_3(s)B}(1+s)}$ (the other cases can be handled similarly)

$$
\begin{aligned}
\bar{\mathcal{K}}^1_{pp}(t,s) \lesssim\ & \gamma_2^2 B \int_{d-\alpha}^{\frac{1}{\sqrt{\gamma_2 cB(1+t)^{1-\frac{\kappa}{2}}}}} \sigma^{3-\frac{1}{\alpha}} e^{-\gamma_2 B\sigma^2(t-s)}\,\mathrm{d}\sigma \\
& + \gamma_2^2 B \int_{\frac{1}{\sqrt{\gamma_2 cB(1+t)^{1-\frac{\kappa}{2}}}}}^{\frac{1}{\sqrt{\gamma_2 cB(1+s)^{1-\frac{\kappa}{2}}}}} \sigma^{3-\frac{1}{\alpha}} e^{-\gamma_2 B\sigma^2(t-s)}(\sigma\sqrt{\gamma_3(t)B}t)^{-(\delta+\frac{\kappa}{2})}\,\mathrm{d}\sigma \\
& + \gamma_2^2 B \int_{\frac{1}{\sqrt{\gamma_2 cB(1+s)^{1-\frac{\kappa}{2}}}}}^{\frac{\sqrt{\gamma_3(t)B}}{\gamma_2 B}} \sigma^{3-\frac{1}{\alpha}} e^{-\gamma_2 B\sigma^2(t-s)}\left(\frac{1+t}{1+s}\right)^{-(\delta+\frac{\kappa}{2})}\,\mathrm{d}\sigma \\
& + \gamma_2^2 B \int_{\frac{\sqrt{\gamma_3(t)B}}{\gamma_2 B}}^{1} \sigma^{3-\frac{1}{\alpha}}\Omega(t)^2 e^{-\int_{\tau_0}^{\tau}\Omega(u)\,\mathrm{d}u}\left(\frac{1+\xi(t)}{1+\xi(s)}\right)^{-2\rho}\,\mathrm{d}\sigma \\
\lesssim\ & \gamma_2^2 B\left((1+\gamma_2 B(t-s))^{-2+\frac{1}{2\alpha}}\left(\frac{1+t}{1+s}\right)^{-\tilde{\delta}} + (\sqrt{\gamma_2 cB}(1+t)^{1-\frac{\kappa}{2}})^{-4+\frac{1}{\alpha}}\right).
\end{aligned}
$$

Here we made the change of variable $u = \sigma\sqrt{\gamma_3(t)B}t$. Notice that the integral behaves as $u^{3-\frac{1}{\alpha}}u^{-(\delta+\kappa/2)}$. It is integrable for $\delta$ large enough, ie $\delta+\kappa/2 > 4-\frac{1}{\alpha}$. We also computed

$$
\begin{aligned}
\int_{\tau_0}^{\tau}(\xi'(u))^2\,\mathrm{d}u &= \int_{\gamma_2 B\sigma^2(1+s)}^{\gamma_2 B\sigma^2(1+t)} \frac{\gamma_3(t(u))}{\gamma_2^2 B\sigma^2}\,\mathrm{d}u \\
&\asymp \gamma_2 B\sigma^2\left[\frac{\gamma_3(t)(1+t)}{\gamma_2^2 B\sigma^2}\right]_s^t
\end{aligned}
$$

where we used $\frac{\mathrm{d}t(u)}{\mathrm{d}u} = \frac{1}{\gamma_2 B\sigma^2}$ since $u = \gamma_2 B\sigma^2 t$.

This allowed to bound the non-exponential term as

$$
\begin{aligned}
\int_{\frac{\sqrt{\gamma_3(t)B}}{\gamma_2 B}}^{1} & \sigma^{3-\frac{1}{\alpha}}\Omega(t)^2 e^{-\int_{\tau_0}^{\tau}\Omega(u)\,\mathrm{d}u}\left(\frac{1+\xi(t)}{1+\xi(s)}\right)^{-2\rho}\,\mathrm{d}\sigma \\
\lesssim\ & \int_{\frac{\sqrt{\gamma_3(t)B}}{\gamma_2 B}}^{1} \sigma^{3-\frac{1}{\alpha}}\left(\frac{\sqrt{\gamma_3(t)B}}{\gamma_2 B\sigma}\right)^4 e^{-(\gamma_3(t)(t-s))/\gamma_2}\left(\frac{1+\xi(t)}{1+\xi(s)}\right)^{-2\rho}\,\mathrm{d}\sigma \\
\lesssim\ & e^{-(\gamma_3(t)t-\gamma_3(s)s)/\gamma_2}\left(\frac{\sqrt{\gamma_3(t)B}}{\gamma_2 B}\right)^4\left(\frac{1+t}{1+s}\right)^{-\tilde{\delta}}\int_{\frac{\sqrt{\gamma_3(t)B}}{\gamma_2 B}}^{1}\sigma^{-1-\frac{1}{\alpha}}\,\mathrm{d}\sigma \\
\lesssim\ & \left(\frac{\sqrt{\gamma_3(t)B}}{\gamma_2 B}\right)^{4-\frac{1}{\alpha}}\left(\frac{1+t}{1+s}\right)^{-\tilde{\delta}} e^{-(\gamma_3(t)(t-s))/\gamma_2} \\
\lesssim\ & (\gamma_2 B(t-s))^{-2+\frac{1}{2\alpha}}\left(\frac{1+t}{1+s}\right)^{-\tilde{\delta}}.
\end{aligned}
$$

Indeed, when $\frac{\gamma_3(t)(t-s)}{\gamma_2} \gtrsim 1$ we know that $e^{-\frac{\gamma_3(t)(t-s)}{\gamma_2}} \lesssim \left(\frac{\gamma_3(t)(t-s)}{\gamma_2}\right)^{-2+\frac{1}{2\alpha}}$ and when $\frac{\gamma_3(t)(t-s)}{\gamma_2} \lesssim 1$ we have $\left(\frac{\sqrt{\gamma_3(t)B}}{\gamma_2 B}\right)^{4-\frac{1}{\alpha}} \lesssim (\gamma_2 B(t-s))^{-2+\frac{1}{2\alpha}}$. Hence the error term is absorbed into the other terms. We additionally used that

$$\left(\frac{1+\xi(t)}{1+\xi(s)}\right)^{-2\rho} \lesssim \left(\frac{1+\xi(t)}{\xi(s)}\right)^{-2\rho} + \left(\frac{1+\xi(t)}{1}\right)^{-2\rho}$$

$$\lesssim \left(\frac{1+t}{1+s}\right)^{-(\delta+\frac{\kappa}{2})} + \xi(t)^{-2\rho}$$

$$\lesssim \left(\frac{1+t}{1+s}\right)^{-(\delta+\frac{\kappa}{2})} + t^{-(1-\kappa_3)2\rho}$$

$$\lesssim \left(\frac{1+t}{1+s}\right)^{-\tilde{\delta}}.$$

where $\tilde{\delta}$ can be chosen arbitrarily large by choosing $\delta$ arbitrarily large.

We proceed similarly for the second term

$$\bar{\mathcal{K}}_{pp}^2(t,s) \lesssim B \int_{d-\alpha}^{\frac{1}{\sqrt{\gamma_2 cB}(1+t)^{1-\frac{\kappa}{2}}}} \sigma^{3-\frac{1}{\alpha}} \frac{\gamma_2 c}{B\sigma^2} e^{-\gamma_2 B\sigma^2(t-s)}(1+s)^{-\kappa}(\sigma\sqrt{\gamma_3(s)B}s)^2 \, d\sigma$$

$$+ B \int_{\frac{1}{\sqrt{\gamma_2 cB}(1+t)^{1-\frac{\kappa}{2}}}}^{\frac{1}{\sqrt{\gamma_2 cB}(1+s)^{1-\frac{\kappa}{2}}}} \sigma^{3-\frac{1}{\alpha}} e^{-\gamma_2 B\sigma^2(t-s)}(\sigma\sqrt{\gamma_3(t)B}t)^{-(\delta+\frac{\kappa}{2})} \frac{\gamma_2 c}{B\sigma^2}(1+s)^{-\kappa}(\sigma\sqrt{\gamma_3(s)B}s)^2 \, d\sigma$$

$$+ B \int_{\frac{1}{\sqrt{\gamma_2 cB}(1+s)^{1-\frac{\kappa}{2}}}}^{1} \sigma^{3-\frac{1}{\alpha}} e^{-\gamma_2 B\sigma^2(t-s)} \left(\frac{1+t}{1+s}\right)^{-(\delta+\frac{\kappa}{2})} \frac{\gamma_2 c}{B\sigma^2}(1+s)^{-\kappa} \, d\sigma$$

$$+ B \int_{\frac{\sqrt{\gamma_3(t)B}}{\gamma_2 B}}^{1} \sigma^{3-\frac{1}{\alpha}} \Omega(t)^2 e^{-\int_{\tau_0}^{\tau} \Omega(u) \, du} \left(\frac{1+\xi(t)}{1+\xi(s)}\right)^{-2\rho} \frac{\gamma_3(s)}{B\sigma^2} \, d\sigma$$

$$\lesssim \gamma_2 c(1+s)^{-\kappa}\left((1+\gamma_2 B(t-s))^{-1+\frac{1}{2\alpha}} \left(\frac{1+t}{1+s}\right)^{-\tilde{\delta}}\right.$$

$$\left. + (\sqrt{\gamma_3(s)B}(1+s))^2(\sqrt{\gamma_3 B}(1+t)^{1-\frac{\kappa}{2}})^{-4+\frac{1}{\alpha}}\right).$$

Here in the second integral we made the change of variable $u = \sqrt{\gamma_3(t)B}t$. We used that $\delta + \kappa/2 > 4 - \frac{1}{\alpha}$ for the integral to converge.

$\square$

**Proposition I.4** (Lower bound on Kernel). *Consider Parametrization I.1 under Assumption 6. Then a lower bound on the kernel function is $\forall t \geq s \geq 0$*

$$\mathcal{K}(t,s) \gtrsim \gamma_2^2 B(\sqrt{\gamma_2 cB}(1+t)^{1-\frac{\kappa}{2}})^{-4+\frac{1}{\alpha}}.$$

*Proof.* Note that $\omega^1(\xi) \asymp 1$ around 0. This gives a lower-bound for $\sigma \lesssim \frac{1}{\sqrt{\gamma_3(t)B}t}$

$$\Phi_{11}^\sigma(t,s) \gtrsim e^{-(\tau(t)-\tau(s))}\left(\frac{1+\xi(t)}{1+\xi(s)}\right)^{-2\rho}$$

$$\gtrsim e^{-\tau(t)}(1+\xi(t))^{-2\rho}.$$

Since $\Phi_{12}^\sigma(t,s)$ is positive, that gives a lower bound on the kernel by integrating

$$\mathcal{K}(t,s) \gtrsim \gamma_2^2 B \int_{d-\alpha}^{\frac{1}{\sqrt{\gamma_2 cB}(1+t)^{1-\frac{\kappa}{2}}}} \sigma^{3-\frac{1}{\alpha}} \times 1 \, d\sigma$$

$$\gtrsim \gamma_2^2 B((\sqrt{\gamma_2 c}(1+t))^{1-\frac{\kappa_3}{2}})^{-4+\frac{1}{\alpha}}.$$

Now just integrate as we did for the forcing function and apply Proposition D.14 since $\sigma \mapsto e^{-\tau(t)}(1 + \xi(t))^{-2\rho}$ is approximately constant in the region $\sigma \lesssim \frac{1}{\sqrt{\gamma_3(t)Bt}}$. $\qquad \square$

**Corollary I.1.** *Let $\vartheta(t) \stackrel{def}{=} 1 + 2\gamma_2 Bt + \left(\int_0^t \sqrt{\gamma_3(s)B}\,\mathrm{d}s\right)^2 \asymp 1 + \left(\int_0^t \sqrt{\gamma_3(s)B}\,\mathrm{d}s\right)^2$. For Parametrization I.1 under Assumption 6, if $\gamma_2 Bt \geq 1$ and $\vartheta(t) \lesssim d^{2\alpha}$, we have*

$$[\mathcal{F} * \mathcal{K}](t) \gtrsim \gamma_2^2 B(\vartheta(t))^{-4+\frac{1}{\alpha}}. \tag{128}$$

*Proof.* Just apply Proposition I.4 and note that $\int_0^t \mathcal{F}(s)\,\mathrm{d}s \gtrsim \int_0^t \mathcal{F}_{pp}(s)\,\mathrm{d}s \gtrsim 1$. $\qquad \square$

**Proposition I.5** (Kesten Lemma). *For Parametrization I.1 under Assumption 6, and for $\tilde{\delta}$ large enough, define*

$$\bar{\bar{\mathcal{K}}}_{pp}(t,s) \stackrel{def}{=} \left(\frac{\gamma_3(s)s}{\gamma_2}\right)^2 \gamma_2^2 B(\sqrt{\gamma_3(t)B}t)^{-4+\frac{1}{\alpha}}$$

$$+ (\gamma_2 B(t-s)^{-1+1/(2\alpha)} \left(\frac{1+t}{1+s}\right)^{-\tilde{\delta}} \left(\gamma_2^2 B \frac{1}{\gamma_2 B(t-s)} + \gamma_3(s)\right).$$

*Then under the assumptions in Proposition I.2 (stability conditions), we have $\forall t \geq s \geq 0$ with $\vartheta(t) \asymp \max\{\gamma_2 Bt, (\sqrt{\gamma_3(t)B}t)^2\} \lesssim d^{2\alpha}$,*

$$\int_{r=s}^t \bar{\bar{\mathcal{K}}}_{pp}(t,r)\bar{\bar{\mathcal{K}}}_{pp}(r,s)\,\mathrm{d}r \leq \epsilon\bar{\bar{\mathcal{K}}}_{pp}(t,s).$$

*Proof.* To prove this upper-bound, the idea is to use that $\bar{\bar{\mathcal{K}}}_{pp}(t,s)$ behaves as a power law and that the stability condition Proposition I.2 ensures that its integral sums at most to one $\int_{s=0}^t \bar{\bar{\mathcal{K}}}_{pp}(t,s)\,\mathrm{d}s \lesssim 1$. For-example we compute for any $t \geq s \geq 0$

$$\int_{r=s}^t \left(\frac{\gamma_3(r)r}{\gamma_2}\right)^2 \gamma_2^2 B(\sqrt{\gamma_3(t)B}t)^{-4+\frac{1}{\alpha}} \times \left(\frac{\gamma_3(s)s}{\gamma_2}\right)^2 \gamma_2^2 B(\sqrt{\gamma_3(t)B}r)^{-4+\frac{1}{\alpha}}\,\mathrm{d}r$$

$$\lesssim \left(\frac{\gamma_3(s)s}{\gamma_2}\right)^2 \gamma_2^2 B(\sqrt{\gamma_3(t)B}t)^{-4+\frac{1}{\alpha}} \times \int_{r=s}^t \left(\frac{\gamma_3(r)r}{\gamma_2}\right)^2 \gamma_2^2 B(\sqrt{\gamma_3(r)B}r)^{-4+\frac{1}{\alpha}}\,\mathrm{d}r$$

$$\lesssim$$

$$\lesssim \left(\frac{\gamma_3(s)s}{\gamma_2}\right)^2 \gamma_2^2 B(\sqrt{\gamma_3(t)B}t)^{-4+\frac{1}{\alpha}}.$$

Here we used that $\left(\frac{\gamma_3(r)r}{\gamma_2}\right)^2 \gamma_2^2 B(\sqrt{\gamma_3(r)B}r)^{-4+\frac{1}{\alpha}} \lesssim r^{2-2\kappa-4+\frac{1}{\alpha}+2\kappa-\frac{\kappa}{\alpha}}$ is integrable for $\kappa = \frac{1}{2\alpha} < 1$.

In the same way we have

$$\int_{r=s}^t ((\gamma_2 B(t-r)^{-1+1/(2\alpha)} \left(\frac{1+t}{1+r}\right)^{-\tilde{\delta}} \left(\gamma_2^2 B \frac{1}{\gamma_2 B(t-r)} + \gamma_3(r)\right)$$

$$\times (\gamma_2 B(r-s)^{-1+1/(2\alpha)} \left(\frac{1+r}{1+s}\right)^{-\tilde{\delta}} \left(\gamma_2^2 B \frac{1}{\gamma_2 B(r-s)} + \gamma_3(s)\right)\mathrm{d}r$$

$$\lesssim \left(\frac{1+t}{1+s}\right)^{-\tilde{\delta}} \times \int_{r=s}^t (\gamma_2 B(t-r)^{-1+1/(2\alpha)}(\gamma_2 B(r-s)^{-1+1/(2\alpha)}$$

$$\times \left(\gamma_2^2 B \frac{1}{\gamma_2 B(t-r)} + \gamma_3(r)\right)\left(\gamma_2^2 B \frac{1}{\gamma_2 B(r-s)} + \gamma_3(s)\right)\mathrm{d}r.$$

There are four possibilities.

We write

$$\int_{r=s}^{t} (\gamma_2 B(t-r))^{-1+1/(2\alpha)} (\gamma_2 B(r-s))^{-1+1/(2\alpha)} \times \gamma_3(r)\gamma_3(s) \, dr$$

$$\lesssim \int_{r=s}^{\frac{s+t}{2}} (\gamma_2 B(t-r))^{-1+1/(2\alpha)} (\gamma_2 B(r-s))^{-1+1/(2\alpha)} \times \gamma_3(r)\gamma_3(s) \, dr$$

$$+ \int_{r=\frac{s+t}{2}}^{t} (\gamma_2 B(t-r))^{-1+1/(2\alpha)} (\gamma_2 B(r-s))^{-1+1/(2\alpha)} \times \gamma_3(r)\gamma_3(s) \, dr$$

$$\lesssim \gamma_3(s)(\gamma_2 B(t-s))^{-1+1/(2\alpha)}.$$

Here we used for the first integral that $\gamma_3(r)(\gamma_2 B(r-s))^{-1+1/(2\alpha)}$ is integrable when $\kappa > \frac{1}{2\alpha}$ and for the second integral that $\gamma_3(r)(\gamma_2 B(t-r))^{-1+1/(2\alpha)}$ is integrable.

Additionally,

$$\int_{r=s}^{t} \gamma_2^2 B(\gamma_2 B(t-r))^{-2+1/(2\alpha)} (\gamma_2 B(r-s))^{-1+1/(2\alpha)} \gamma_3(s) \, dr$$

$$\lesssim \int_{r=s}^{\frac{s+t}{2}} \gamma_2^2 B(\gamma_2 B(t-r))^{-2+1/(2\alpha)} (\gamma_2 B(r-s))^{-1+1/(2\alpha)} \gamma_3(s) \, dr$$

$$+ \int_{r=\frac{s+t}{2}}^{t} \gamma_2^2 B(\gamma_2 B(t-r))^{-2+1/(2\alpha)} (\gamma_2 B(r-s))^{-1+1/(2\alpha)} \gamma_3(s) \, dr$$

$$\lesssim \gamma_3(s)(\gamma_2 B(t-s))^{-1+1/(2\alpha)}.$$

Here we used for the first integral that $\int_{r=s}^{\frac{s+t}{2}} \gamma_2^2 B(\gamma_2 B(t-r))^{-2+1/(2\alpha)} \, dr \lesssim \gamma_2(\gamma_2 B(t-s))^{-1+1/(2\alpha)}$ and for the second integral that $\int_{r=\frac{s+t}{2}}^{t} \gamma_2^2 B(\gamma_2 B(t-r))^{-2+1/(2\alpha)} \, dr \lesssim \gamma_2(\gamma_2 B(t-s))^{-1+1/(2\alpha)} \lesssim 1$.

Similarly,

$$\int_{r=s}^{t} (\gamma_2 B(t-r))^{-1+1/(2\alpha)} \gamma_3(r)(\gamma_2 B(r-s))^{-2+1/(2\alpha)} \gamma_2^2 B \, dr$$

$$\lesssim \int_{r=s}^{\frac{s+t}{2}} (\gamma_2 B(t-r))^{-1+1/(2\alpha)} \gamma_3(r)(\gamma_2 B(r-s))^{-2+1/(2\alpha)} \gamma_2^2 B \, dr$$

$$+ \int_{r=\frac{s+t}{2}}^{t} (\gamma_2 B(t-r))^{-1+1/(2\alpha)} \gamma_3(r)(\gamma_2 B(r-s))^{-2+1/(2\alpha)} \gamma_2^2 B \, dr$$

$$\lesssim \gamma_3(s)(\gamma_2 B(t-s))^{-1+1/(2\alpha)} + \gamma_2^2 B(\gamma_2 B(t-s))^{-2+\frac{1}{2\alpha}}.$$

We used in the first integral that $\gamma_3(r) \lesssim \gamma_3(s)$ and that $(\gamma_2 B(r-s))^{-2+\frac{1}{2\alpha}}$ integrable. We used in the second integral that $(\gamma_2 B(t-r))^{-1+1/(2\alpha)} \gamma_3(r)$ is integrable.

Finally it is clear since $(\gamma_2 B(t-r))^{-2+\frac{1}{2\alpha}}$ is integrable that

$$\int_{r=s}^{t} \gamma_2^2 B(\gamma_2 B(t-r))^{-2+1/(2\alpha)} \gamma_2^2 B(\gamma_2 B(r-s))^{-2+1/(2\alpha)} \, dr$$

$$= \int_{r=s}^{\frac{s+t}{2}} \gamma_2^2 B(\gamma_2 B(t-r))^{-2+1/(2\alpha)} \gamma_2^2 B(\gamma_2 B(r-s))^{-2+1/(2\alpha)} \, dr$$

$$+ \int_{r=\frac{s+t}{2}}^{t} \gamma_2^2 B(\gamma_2 B(t-r))^{-2+1/(2\alpha)} \gamma_2^2 B(\gamma_2 B(r-s))^{-2+1/(2\alpha)} \, dr$$

$$\lesssim \gamma_2^2 B(\gamma_2 B(t-s))^{-2+1/(2\alpha)}.$$

There remains to bound

$$\int_{r=s}^{t} \left(\frac{\gamma_3(r)r}{\gamma_2}\right)^2 \gamma_2^2 B(\sqrt{\gamma_3(t)B}t)^{-4+\frac{1}{\alpha}}$$

$$\times (\gamma_2 B(r-s))^{-1+1/(2\alpha)} \left(\frac{1+r}{1+s}\right)^{-\tilde{\delta}} \left(\gamma_2^2 B \frac{1}{\gamma_2 B(r-s)} + \gamma_3(s)\right) dr$$

$$\lesssim \left(\frac{\gamma_3(s)s}{\gamma_2}\right)^2 \gamma_2^2 B(\sqrt{\gamma_3(t)B}t)^{-4+\frac{1}{\alpha}}$$

$$\times \int_{r=s}^{t} (\gamma_2 B(r-s))^{-1+1/(2\alpha)} \left(\frac{1+r}{1+s}\right)^{-\tilde{\delta}+2(1+\kappa)} \left(\gamma_2^2 B \frac{1}{\gamma_2 B(r-s)} + \gamma_3(s)\right) dr$$

$$\lesssim \left(\frac{\gamma_3(s)s}{\gamma_2}\right)^2 \gamma_2^2 B(\sqrt{\gamma_3(t)B}t)^{-4+\frac{1}{\alpha}} \left(\int_{r=s}^{t} \gamma_2^2 B(\gamma_2 B(r-s))^{-2+1/(2\alpha)} dr\right.$$

$$\left. + \int_{r=s}^{t} \gamma_3(s)(\gamma_2 B(r-s))^{-1+1/(2\alpha)} \left(\frac{1+r}{1+s}\right)^{-\tilde{\delta}+2(1+\kappa)} dr\right)$$

$$\lesssim \left(\frac{\gamma_3(s)s}{\gamma_2}\right)^2 \gamma_2^2 B(\sqrt{\gamma_3(t)B}t)^{-4+\frac{1}{\alpha}}.$$

Here we used for the first term that $(\gamma_2 B(r-s))^{-2+\frac{1}{2\alpha}}$ is integrable and for the second term that $-\tilde{\delta} + 2(1+\kappa) < -\kappa$ so that $(\gamma_2 B(r-s)^{-1+\frac{1}{2\alpha}} \times (1+r)^{-\kappa}$ is integrable since $\kappa > \frac{1}{2\alpha}$ and that $\gamma_3(s)(1+s)^\kappa \lesssim 1$.

Finally, we need to bound the symmetric term i.e.

$$\int_{r=s}^{t} \left(\frac{\gamma_3(s)s}{\gamma_2}\right)^2 \gamma_2^2 B(\sqrt{\gamma_3(r)B}r)^{-4+\frac{1}{\alpha}}$$

$$\times (\gamma_2 B(t-r))^{-1+1/(2\alpha)} \left(\frac{1+t}{1+r}\right)^{-\tilde{\delta}} \left(\gamma_2^2 B \frac{1}{\gamma_2 B(t-r)} + \gamma_3(r)\right) dr$$

$$\lesssim \left(\frac{\gamma_3(s)s}{\gamma_2}\right)^2 \gamma_2^2 B(\sqrt{\gamma_3(t)B}t)^{-4+\frac{1}{\alpha}}$$

$$\times \int_{r=s}^{t} (\gamma_2 B(t-r))^{-1+1/(2\alpha)} \left(\frac{1+t}{1+r}\right)^{-\tilde{\delta}+4-\frac{1}{\alpha}} \left(\gamma_2^2 B \frac{1}{\gamma_2 B(t-r)} + \gamma_3(r)\right) dr$$

$$\lesssim \left(\frac{\gamma_3(s)s}{\gamma_2}\right)^2 \gamma_2^2 B(\sqrt{\gamma_3(t)B}t)^{-4+\frac{1}{\alpha}}.$$

Here we used that $-(\delta + \frac{\kappa}{2}) + 4 - \frac{1}{\alpha} < 0$ and that

$$\int_{r=s}^{t} (\gamma_2 B(t-r))^{-1+1/(2\alpha)} \left(\gamma_2^2 B \frac{1}{\gamma_2 B(t-r)} + \gamma_3(r)\right) dr \lesssim 1.$$

$\square$

**Proposition I.6** (Upper-bound on kernel contribution)**.** *Denote for $\tilde{\delta}$ large enough*

$$\bar{\bar{\mathcal{K}}}_{pp}(t,s) \stackrel{def}{=} \left(\frac{\gamma_3(s)s}{\gamma_2}\right)^2 \gamma_2^2 B(\sqrt{\gamma_3(t)B}t)^{-4+\frac{1}{\alpha}}$$

$$+ (\gamma_2 B(t-s)^{-1+1/(2\alpha)} \left(\frac{1+t}{1+s}\right)^{-\tilde{\delta}} \left(\gamma_2^2 B \frac{1}{\gamma_2 B(t-s)} + \gamma_3(s)\right).$$

*Then if $\gamma_2 \asymp 1$ and $\gamma_3 \asymp 1$ and under stability in Proposition I.2 we have $\forall t \geq s \geq 0$ with $\vartheta(t) \lesssim d^{2\alpha}$*

$$[\mathcal{F} * \bar{\bar{\mathcal{K}}}_{pp}](t) \lesssim \mathcal{F}(t) + \bar{\bar{\mathcal{K}}}_{pp}(t).$$

*Proof.* $\bar{\bar{\mathcal{K}}}_{pp}(t,s)$ has two main terms we treat separately. For the first contribution of the first term, there are two cases, whether $\mathcal{F}(t) \times \left(\frac{\gamma_3(t)t}{\gamma_2}\right)^2$ is integrable or not.

**Not integrable** Then we check that

$$\int_0^t \mathcal{F}(s) \left(\frac{\gamma_3(s)s}{\gamma_2}\right)^2 \gamma_2^2 B(\sqrt{\gamma_3(t)B}(1+t))^{-4+\frac{1}{\alpha}} \, ds$$

$$\lesssim \mathcal{F}(t) \times \left(\frac{\gamma_3(t)t}{\gamma_2}\right)^2 (1+t)\gamma_2^2 B(\sqrt{\gamma_3(t)B}(1+t))^{-4+\frac{1}{\alpha}}$$

$$\lesssim \mathcal{F}(t).$$

Here we used that since $\alpha > \frac{1}{2}$

$$\left(\frac{\gamma_3(t)t}{\gamma_2}\right)^2 (1+t)\gamma_2^2 B(\sqrt{\gamma_3(t)B}(1+t))^{-4+\frac{1}{\alpha}}$$

$$\asymp (1+t)^{-1+\frac{1}{\alpha}-\frac{1}{4\alpha^2}}$$

$$\lesssim 1.$$

**Integrable** Then we write

$$\int_0^t \mathcal{F}(s) \left(\frac{\gamma_3(s)s}{\gamma_2}\right)^2 \gamma_2^2 B(\sqrt{\gamma_3(t)B}(1+t))^{-4+\frac{1}{\alpha}} \, ds$$

$$\lesssim \int_0^t \mathcal{F}(s) \times \left(\frac{\gamma_3(s)s}{\gamma_2}\right)^2 ds\gamma_2^2 B(\sqrt{\gamma_3(t)B}(1+t))^{-4+\frac{1}{\alpha}}$$

$$\lesssim \bar{\bar{\mathcal{K}}}_{pp}(t,0).$$

For the contribution of the second term, just notice that since $\tilde{\delta}$ is large enough, $\mathcal{F}(s)(1+s)^\delta$ is not integrable. By cutting the integral in two pieces and noticing that $\int_0^t \bar{\bar{\mathcal{K}}}_{pp}(t,s) \, ds < 1$, we obtain that the contribution is smaller than $\mathcal{F}(t)$. This concludes the proof.

$\square$

**Theorem I.2** (Scaling Law for DANA-decaying)**.** *Let $\alpha > \frac{1}{2}$ and $B \asymp 1$. Consider Parametrization I.1 under Assumption 6. Suppose that $\delta$ is large enough, ie $\delta > \bar{\delta}(\kappa, \alpha, \beta)$,* [13] *Then for any $\epsilon > 0$ there exists $\mathfrak{g}(\kappa, \epsilon) > 0$, $C > 0$ and $d_0$ large enough, such that for any $d \geq d_0$ if $\gamma_2 = \mathfrak{g}$ and $c \leq \mathfrak{g}$ we have*

$$\frac{1}{C}(\mathcal{F}_0 + \mathcal{F}_{ac}(t) + \mathcal{F}_{pp}(t) + \mathcal{K}_{pp}(t)) \lesssim \mathcal{P}(t)$$

$$\leq C(\mathcal{F}_0 + \mathcal{F}_{ac}(t) + \mathcal{F}_{pp}(t) + \mathcal{K}_{pp}(t))$$

*where $\mathcal{F}_0, \mathcal{F}_{pp}, \mathcal{F}_{ac}$ have the asymptotics given in Proposition I.1*

$$\mathcal{K}_{pp}(t) = \gamma_2^2 B(1 + \sqrt{\gamma_3(t)B}t)^{-4+\frac{1}{\alpha}}.$$

---

[13]We believe $\bar{\delta}$ to behave at least such that $\frac{\bar{\delta}+\frac{\kappa}{2}}{1-\frac{\kappa}{2}} > \max\{\frac{2\alpha+2\beta-1}{\alpha}, 4-\frac{1}{\alpha}\}$, and $(\bar{\delta}+\kappa/2) > 2+3\kappa$

| **General learning rate schedule, $\gamma_3(t)$** | **Specific learning rate schedule,** $\gamma_3(t) \sim (1+t)^{-1/(2\alpha)}$ |
|---|---|
| $\mathcal{F}_0(t) \asymp d^{-2\alpha+\max\{0,1-2\beta\}}$ | $\mathcal{F}_0(t) \asymp d^{-2\alpha+\max\{0,1-2\beta\}}$ |
| $\mathcal{F}_{pp}(t) \asymp \hat{\mathcal{F}}_{pp}(\vartheta(t))$ | $\mathcal{F}_{pp}(t) \asymp \begin{cases} (\gamma_2 Bt)^{-1-\frac{2\beta-1}{2\alpha}}, & \text{if } 2\alpha < 1 \\ (\tau(t)^2)^{-1-\frac{2\beta-1}{2\alpha}}, & \text{if } 2\alpha > 1 \end{cases}$ |
| $\mathcal{F}_{ac}(t) \lesssim \begin{cases} \mathcal{F}_0(t), \text{ if } 2\beta > 1, 2\alpha < 1 \\ 0, \text{ if } 2\beta < 1 \end{cases}$ 
 if $2\beta > 1, 2\alpha > 1$, 
 $\mathcal{F}_{ac}(t) \asymp \hat{\mathcal{F}}_{ac}(\vartheta(t))$ | $\mathcal{F}_{ac}(t) \lesssim \begin{cases} \mathcal{F}_0(t), \text{ if } 2\beta > 1, 2\alpha > 1 \\ 0, \text{ if } 2\beta < 1 \end{cases}$ 
 if $2\beta > 1, 2\alpha > 1$, $\mathcal{F}_{ac} \asymp (\tau(t))^{-2+1/(\alpha)} d^{-1}$ |
| $\mathcal{K}_{pp}(t,0) \asymp \gamma_2^2 B\hat{\mathcal{K}}_{pp}(\vartheta(t))$ | $\mathcal{K}_{pp}(t,0) \asymp \begin{cases} B\gamma_2^2(\gamma_2 Bt)^{-2+1/(2\alpha)}, & \text{if } 2\alpha < 1 \\ B\gamma_2^2(\tau(t)^2)^{-2+1/(2\alpha)}, & \text{if } 2\alpha > 1 \end{cases}$ |

Table 12: **DANA-decaying: Large $d$ behavior of the forcing function and kernel function for general and specific $\gamma_3$ schedules.** See Section B.4 for details about the algorithm. The constant $C$ is independent of dimension and the function $\tau(t,s) \stackrel{\text{def}}{=} \int_s^t \sqrt{\gamma_3(u)B}\,\mathrm{d}u$ with $\tau(t) \stackrel{\text{def}}{=} \tau(t,0)$ and we remind from Theorem 4.1 that $\vartheta(t) \asymp 1 + \gamma_2 Bt + \tau(t)^2$. In the case where $\gamma_3(t) \sim (1+t)^{-1/(2\alpha)}$, $\tau(t,0) \sim B(1+t)^{1-1/(4\alpha)}$. The definitions of $\hat{\mathcal{F}}_i$ and $\hat{\mathcal{K}}_{pp}$ can be found in the introduction.

*Proof.* We know that the solution of the Volterra equation can be written by repeated convolution of the forcing with kernel function, i.e. $\forall t \geq 0$

$$\mathcal{P}(t) = \mathcal{F}(t) + \sum_{k=1}^{\infty}[\mathcal{F} * \mathcal{K}^{*k}](t).$$

For the lower-bound, we apply Proposition I.1 and Corollary I.1. For the upper-bound we apply Lemma C.3 with Kesten's Lemma Proposition I.5 on the upper-bound $\bar{\bar{\mathcal{K}}}_{pp}$ on the kernel (Proposition I.3) with the upper-bound on $\mathcal{F} * \mathcal{K}$ from Proposition I.6. $\qquad\square$

### I.5 Extended heuristics for general algorithm

In the following we discuss heuristics to justify the scaling laws of the general (DANA) algorithm that are formulated in Theorem 4.1 and more precisely (10).

**Claim I.1.** *Denote* $\gamma_1(t) \equiv 1$, $\gamma_2(t) = \tilde{\gamma}_2 d^{-\kappa_1}$, $\gamma_3(t) = \tilde{\gamma}_3 d^{-\kappa_2}(1+t)^{-\kappa_3}$, $\Delta(t) = \delta(1+t)^{-1}$ *and* $B = c_b d^{\kappa_b}$. *Denote* $\rho \stackrel{\text{def}}{=} \frac{\frac{\delta}{2}+\frac{\kappa_3}{4}}{1-\frac{\kappa_3}{2}}$.

*For* $\alpha > \frac{1}{2}$ *take* $2\rho > \max\{\frac{2\alpha+2\beta-1}{\alpha}, 4-\frac{1}{\alpha}\}$ *and* $2\rho(1-\kappa_3/2) > 1$.

- $\kappa_3 \geq 1$, $\eta_2 = \eta_3 = 0$. *Same scaling laws as SGD.*

- $1 \geq \kappa_3 \geq \frac{1}{2\alpha}$, $\eta_2 = \eta_3 = 0$. *Scaling laws in-between DANA-decaying and SGD.*

- $0 \leq \kappa_3 < \frac{1}{2\alpha}$, $\eta_2 = 0$, $\eta_3 = 2\alpha(\frac{1}{2\alpha} - \kappa_3)$. *Scaling laws in-between DANA-constant and DANA-decaying.*

*The scaling laws are given by* (10).

*If* $2\rho < \max\{\frac{2\alpha+2\beta-1}{\alpha}, 4-\frac{1}{\alpha}\}$ *then the exponent in* $\mathcal{F}_{pp}, \mathcal{F}_{ac}, \mathcal{K}_{pp}$ *is replaced by the minimum between its exponent and $\rho$ plus SGD exponent after it started accelerating.*

### I.5.1 Forcing function

**Claim I.2** (Forcing function). *Let $\alpha > 0$, $2\alpha + 2\beta > 1$, $\alpha, \beta \neq \frac{1}{2}$, $\alpha + 1 > \beta$. Suppose that $2\rho \overset{def}{=} \frac{\delta + \frac{1}{4\alpha}}{1 - \frac{1}{4\alpha}} > \frac{2\alpha + 2\beta - 1}{\alpha}$. There exists some $C(\alpha, \beta > 0$ such that under Parametrization I.1, denote $\vartheta(t) \overset{def}{=} 1 + 2\gamma_2 Bt + \left( \int_0^t \sqrt{\gamma_3(s)B} \, ds \right)^2$ we have for any $t \geq 0$ and $d$ large enough*

$$\frac{1}{C} \left( \mathcal{F}_0(t) + \mathcal{F}_{pp}(t) + \mathcal{F}_{ac}(t) \right) \leq \mathcal{F}(t) \leq C \left( \mathcal{F}_0(t) + \mathcal{F}_{pp}(t) + \mathcal{F}_{ac}(t) \right)$$

*where*

$$\mathcal{F}_0 \asymp d^{-2\alpha + (1 - 2\beta)_+},$$

$$\mathcal{F}_{pp}(t) \asymp \vartheta(t)^{-1 - \frac{2\beta - 1}{2\alpha}},$$

$$\mathcal{F}_{ac}(t) \begin{cases} \asymp \vartheta(t)^{-1 + \frac{1}{2\alpha}} \text{ if } 2\alpha > 1, \ 2\beta > 1 \\ \lesssim \mathcal{F}_0 \text{ if } 2\alpha < 1, \ 2\beta > 1 \\ 0 \text{ else.} \end{cases}$$

*Idea:* We define $\mathcal{F}_0, \mathcal{F}_{pp}, \mathcal{F}_{ac}$ as in Section D and use the estimate on $\Phi_{11}^{\sigma^2}(t, 0)$ from Theorem I.1. $\mathcal{F}_0$ is unchanged, however for example for $\mathcal{F}_{pp}(t)$ we compute

$$\mathcal{F}_{pp}(t) \overset{def}{=} \int_0^1 \sigma^{1 + \frac{2\beta - 1}{\alpha}} \left( e^{-\gamma_2 B\sigma^2(1+t)} (1 + \frac{\sqrt{\gamma_2 B\sigma^2}}{1 - \frac{\kappa}{2}} ((1+t)^{1 - \frac{\kappa}{2}} - 1))^{-2\rho} \right.$$

$$\left. + \mathcal{O}(\epsilon (\gamma_2 B\sigma^2)^{\delta + \frac{\kappa}{2}} e^{-\mathfrak{c}\gamma_2 B\sigma^2}) \right)$$

$$\asymp \int_{\sigma = 0}^{\min\{ \frac{1}{\sqrt{\gamma_2 Bt}}, \frac{1}{\sqrt{\gamma_2 Bc(1+t)^{1 - \frac{\kappa}{2}}}}, 1 \}} \sigma^{1 + \frac{2\beta - 1}{\alpha}} \, d\sigma$$

$$\asymp \vartheta(t)^{-1 - \frac{2\beta - 1}{2\alpha}}.$$

Where we used that $-2\rho + 1 - \frac{2\beta - 1}{\alpha} < -1 \iff 2\rho > \frac{2\alpha + 2\beta - 1}{\alpha}$. Similar computation on $\mathcal{F}_{ac}$ brings the result, with the additional condition $2\rho > 2 - \frac{1}{\alpha}$ which is automatically satisfied since $\frac{2\alpha + 2\beta - 1}{\alpha} > 2 - \frac{1}{\alpha}$.

Finally, to bound $\mathcal{F}$ using $\mathcal{F}_0(t) + \mathcal{F}_{pp}(t) + \mathcal{F}_{ac}(t)$ we proceed with a similar proof as Proposition H.16 by noting that in the range of interest (last integral), $\Phi_{11}^{\sigma^2}(t, 0)$ satisfies the hypothesis in Proposition D.13 (it is constant).

### I.6 Stability conditions

**Claim I.3** (Necessary and sufficient conditions for stability above the high-dimensional line $\alpha > \frac{1}{2}$ and with batch $B = 1$). *Let $\alpha > \frac{1}{2}$. Consider Parametrization I.1 with $B = 1$, $\kappa \geq 0$, $\gamma_2 > 0$. Suppose that $2\rho > \max\{ \frac{2\alpha + 2\beta - 1}{\alpha}, 4 - \frac{1}{\alpha} \}$. We have*

- *(Sufficient condition) For any $\epsilon > 0$ there exists $\mathfrak{g}(\kappa, \epsilon) > 0$ and $d_0$ large enough, such that for any $d \geq d_0$ if $(\kappa > \frac{1}{2\alpha}, \gamma_2 = \mathfrak{g}$ and $c \leq \mathfrak{g})$ or $(\kappa < \frac{1}{2\alpha}, \gamma_2 = \mathfrak{g}$ and $c \leq \mathfrak{g}d^{2\alpha(\kappa - \frac{1}{2\alpha})})$ then $\sup_{t \geq 0} \int_0^t \mathcal{K}(t, s) \, ds < \epsilon$. In particular, since $\mathcal{F}(t)$ (is bounded indep of $d$), we have $\sup_{d \geq d_0} \|\mathcal{P}\|_\infty < \infty$.*

- *(Necessary condition) For any $\epsilon > 0$, any $\mathfrak{g} > 0$ and any $d_0 \in \mathbb{N}$ large enough, for any $d \geq d_0$ and if $(\kappa < \frac{1}{2\alpha}, \gamma_2 = \mathfrak{g}$ and $c = \mathfrak{g}d^{\tilde{\kappa}}$ with $0 > \tilde{\kappa} > 2\alpha(\kappa - \frac{1}{2\alpha}))$, then there exists $\sigma_1 > 0$, $\sigma_2 > 0$ such that for any $d^{2\alpha} > t \geq d^{2\alpha - \sigma_1}$ we have $\int_{t/2}^t \mathcal{K}(t, s) \, ds > d^{\sigma_2}$. In particular under this scaling $\sup_{d \geq d_0} \|\mathcal{P}\|_\infty = \infty$*

- *For any $\epsilon > 0$ and $\kappa > 0$, there exists some $\mathfrak{g} > 0$ such that if $\gamma_2 \leq \mathfrak{g}$ then for $d_0$ large enough and any $d \geq d_0$, $\limsup_{t\to\infty} \int_0^t \mathcal{K}(t,s)\,ds \leq \epsilon$ and as a consequence, $\|\mathcal{P}\|_\infty < \infty$.*

*Idea: Sufficient condition* We again consider the estimates in Theorem I.1. The goal is to have sufficient conditions so that $\int_0^t \mathcal{K}(t,s)\,ds < \epsilon$ for any $t \geq 0$.

We compute the first term by bounding the oscillatory $\omega^1(\xi, \xi_0), \omega^2(\xi, \xi_0)$ by constants and $\frac{1+\xi}{1+\xi_0} \leq 1$. For $M > 0$ large,

$$\int_0^t \mathcal{K}^1(t,s)\,ds \lesssim \gamma_2^2 B \int_0^t \int_{\sigma=\frac{1}{M}d^{-\alpha}}^1 \sigma^{3-\frac{1}{\alpha}} e^{-\gamma_2 B\sigma^2(t-s)}\,d\sigma\,ds$$

$$\lesssim \gamma_2 \int_{\sigma=\frac{1}{M}d^{-\alpha}}^1 \sigma^{1-\frac{1}{\alpha}}[1 - e^{-\gamma_2 B\sigma^2 t}]\,d\sigma$$

$$\lesssim \gamma_2 d^{(1-2\alpha)_+}.$$

For the second term we do the same and use that $(\gamma_2 \xi'(\tau(s)))^2 \asymp \frac{\gamma_3}{B\sigma^2}(1+s)^{-\kappa}$ to write for $\gamma_2 B(1+t) \lesssim d^{2\alpha}$

$$\int_0^t \mathcal{K}^2(t,s)\,ds \lesssim B \int_0^t \int_{\sigma=\frac{1}{M}d^{-\alpha}}^1 \sigma^{3-\frac{1}{\alpha}}\frac{\gamma_3}{B\sigma^2}(1+s)^{-\kappa}e^{-\gamma_2 B\sigma^2(t-s)}\,d\sigma\,ds$$

$$\lesssim \gamma_3 \int_{\sigma=\frac{1}{M}d^{-\alpha}}^1 \sigma^{1-\frac{1}{\alpha}}\min\{(1+t)^{-\kappa}\frac{1}{\gamma_2 B\sigma^2}, (1+t)^{-\kappa+1}\}\,d\sigma$$

$$\lesssim \gamma_3 \int_{\sigma=\frac{1}{M}d^{-\alpha}}^{\frac{1}{\sqrt{\gamma_2 B(1+t)}}} \sigma^{1-\frac{1}{\alpha}}(1+t)^{-\kappa+1}\,d\sigma + \gamma_3 \int_{\frac{1}{\sqrt{\gamma_2 B(1+t)}}}^1 (1+t)^{-\kappa}\frac{\sigma^{-1-\frac{1}{\alpha}}}{\gamma_2 B}\,d\sigma$$

$$\lesssim \gamma_3(1+t)^{-\kappa+1}(\gamma_2 B(1+t))^{-1+\frac{1}{2\alpha}} + \frac{\gamma_3}{\gamma_2 B}(1+t)^{-\kappa}(\gamma_2 B(1+t))^{\frac{1}{2\alpha}}.$$

When $\gamma_2 B(1+t) \gtrsim d^{2\alpha}$ the first term vanishes and we obtain

$$\int_0^t \mathcal{K}^2(t,s)\,ds \lesssim \frac{\gamma_3}{\gamma_2 B}(1+t)^{-\kappa}\int_{\frac{1}{M}d^{-\alpha}}^1 \sigma^{-1-\frac{1}{\alpha}}\,d\sigma$$

$$\lesssim \frac{\gamma_3}{\gamma_2 B}(1+t)^{-\kappa} \times d.$$

Evaluating at the worst case $t \asymp \frac{d^{2\alpha}}{\gamma_2 B}$ yields the stability condition

$$\frac{\gamma_3}{\gamma_2 B}\Big(\frac{d^{2\alpha}}{\gamma_2 B}\Big)^{-\kappa} \times d \lesssim 1. \tag{129}$$

*Necessary condition* To obtain the necessary condition, just observe that the previous upper-bounds are in fact tights. Indeed, for $\xi_0 \asymp \xi$, the term $\left(\frac{1+\xi}{1+\xi_0}\right)^{-\delta}$ can be treated as constant. Since we additionally know lower-bounds on $\omega^1(\xi), \omega^2(\xi)$ we deduce the corresponding lower bound on $\mathcal{K}^1(t,s), \mathcal{K}^2(t,s)$. This brings that for $t \lesssim d^{2\alpha}$

$$\int_{t/2}^t \mathcal{K}(t,s)\,ds \gtrsim \gamma_3(t)(\gamma_2 Bt)^{1/(2\alpha)} \gtrsim d^{\tilde\kappa}(1+t)^{-\kappa+\frac{1}{2\alpha}}.$$

Since by assumption, $\tilde\kappa + 2\alpha(-\kappa + \frac{1}{2\alpha}) > 0$ we obtain the existence of $\sigma_1 > 0$, $\sigma_2 > 0$ such that $\forall t \geq d^{2\alpha-\sigma_1}$ we have $\int_{t/2}^t \mathcal{K}(t,s)\,ds \geq d^{\sigma_2}$. Now we know that $\forall t \geq 0$, $\forall \alpha > 0$, $\mathcal{F}(t) \gtrsim d^{-2\alpha}$.

After $t \gtrsim d^{2\alpha-\sigma_1}$ we hence obtain by recursion that the loss grows as $d^{\sigma_2})^{\log_2(d)}$. This brings that $\mathcal{P}(d^{2\alpha}) \gtrsim d^{-2\alpha} d^{\sigma_2 \sigma_1 \log_2(d)}$. We see that this diverges as $d \to \infty$ which concludes.

*Third point* The last point in the proposition states that even when the conditions for stability are broken, the loss remains bounded (even though it can increase arbitrarily with dimension for times $t \lesssim d^{2\alpha}$. To see it just observe as before that $\int_0^t \mathcal{K}^1(t,s)\,\mathrm{d}s \lesssim \gamma_2$ and that for $\gamma_2 Bt \gtrsim d^{2\alpha}$, we have $\int_0^t \mathcal{K}^2(t,s)\,\mathrm{d}s \lesssim (1+t)^{-\kappa}\frac{\gamma_3}{\gamma_2 B} \times d \to 0$.

## I.7  Kernel function

Now that we have the stability conditions we can compute the kernel function. We will focus on the contribution from $\Phi_{11}^\sigma(t,s)$ which we remind the behavior from Theorem I.1

$$\Phi_{11}(t,s) \asymp e^{-(\tau(t)-\tau(s))} \left(\frac{1+\xi(t)}{1+\xi(s)}\right)^{-2\rho}.$$

Then we can give the behavior of $\mathcal{K}^1(t,0)$.

**Claim I.4.** *If $\vartheta(t) \lesssim d^{2\alpha}$ we have*

$$\mathcal{K}^1(t,0) \asymp \gamma_2^2 B\left((\gamma_2 Bt)^{-2+\frac{1}{2\alpha}} + (\sqrt{\gamma_3(t)B}(1+t))^{-4+\frac{1}{\alpha}}\right).$$

*Idea:* As for the forcing function, there are two cases depending on which contribution from $\gamma_2 Bt$ or $(\sqrt{\gamma_3(t)B}t)^2$ is dominant in $\vartheta(t)$. Hence we decompose

$$\begin{aligned}
\mathcal{K}^1(t,s) &\asymp \gamma_2^2 B \int_{d^{-\alpha}}^{\min\{\frac{1}{\gamma_2 Bt}, \frac{1}{\sqrt{\gamma_3(t)B}t}\}} e^{-(\tau(t)-\tau(s))}(1+\xi(t))^{-2\rho}\,\mathrm{d}\sigma \\
&\asymp \gamma_2^2 B\vartheta(t)^{-2+\frac{1}{2\alpha}}.
\end{aligned}$$

## J  Compute-optimality beyond stability and motivation for DANA-decaying schedule

In the previous sections, we restricted the learning rates domain to describe a *stable* algorithm, i.e. requiring that the risk $\mathcal{P}(t)$ stays bounded for all time $t \in (0,\infty)$. In this section, we ask and heuristically answer the following question:

Does there exist for classical momentum (equiv. SGD) or DANA-constant a scaling in $d$ of the learning rates $\gamma_2, \gamma_3$ which yields a *better compute-optimal frontier* without requiring the algorithm to be stable for any time $t \in \mathbb{R}_+$?

### J.1  Strategy

In the previous study, we imposed stability of the algorithm by controlling the kernel norm $\forall t \geq 0$, $\int_0^t \mathcal{K}(t,s)\,\mathrm{d}s < 1$. This, combined with a corresponding Kesten's lemma (see [80, Lem. C.1] for SGD, Lemma F.3 for classical momentum, Lemma C.2 for DANA-constant) ensures that the resolvent $r(t,s) \stackrel{\text{def}}{=} \sum_{k=1}^\infty \mathcal{K}^{*k}(t,s)$ of the Volterra equation (55) is bounded for all time and leveraging [45] implies that the solution $\mathcal{P}(t) = \mathcal{F}(t) + [\mathcal{F} * r](t)$ is bounded.

Instead it is clear by following the same steps, that if for some $T > 0$, $\forall t \leq T$, $\int_0^t \mathcal{K}(t,s)\,\mathrm{d}s < 1$, we can recover that $\mathcal{P}(t)$ stays bounded for $t \leq T$. More precisely, retracing the same steps in the proofs that give rise to Eq. (62) (see Theorem G.1 for SGD-M, see Theorem H.3 for DANA-constant), we would obtain $\forall t \leq T$, $\mathcal{P}(t) \asymp \mathcal{F}(t) + \frac{1}{\gamma_2 B}\mathcal{K}(t,0)$. On the other hand, if for $t > T$, $\int_0^t \mathcal{K}(t,s)\,\mathrm{d}s > 1$ then, starting $t \geq T$, $\mathcal{P}(t)$ will start diverging exponentially. Taking $T = +\infty$ (or equivalently $d^{2\alpha}/(\gamma_2 B)$ for SGD and $\min\{d^{2\alpha}/(\gamma_2 B), d^\alpha/\sqrt{\gamma_3 B}\}$ for DANA-constant) recovers the previous study.

| | $\kappa$ | Largest $\gamma_2$ stable at compute-optimality | Compute-optimal $\gamma_2$ |
|---|---|---|---|
| **Phases Ia, II, III** | Any | $\gamma_2 \asymp 1$ | $\gamma_2 \asymp 1$ |
| **Phase Ib** | $\kappa = 2\alpha$ | $\gamma_2 \asymp d^{2\alpha-1}$ | $\gamma_2 \asymp d^{2\alpha-1}$ |
| **Phase Ic** | $\kappa = \frac{4\alpha^2}{2\alpha+2\beta-1}$ | $\gamma_2 \asymp d^{\frac{2\alpha(2\alpha-1)}{2\alpha+2\beta-1}}$ | $\gamma_2 \asymp d^{\frac{2\alpha(2\alpha-1)}{2\alpha+2\beta-1}}$ |
| **Phase IVa** | $2\alpha$ | $\gamma_2 \asymp d^{2\alpha-1}$ | $\gamma_2 \asymp d^{-\frac{4\alpha^2-4\alpha\beta}{2\alpha+2\beta-1}}$ (cf [80]) |
| **Phase IVb** | $\kappa = \frac{\alpha(2\alpha-1)}{\alpha-\beta}$ | $\gamma_2 \asymp d^{\frac{(2\alpha-1)^2}{2(\alpha-\beta)}} \gg d^{2\alpha-1}$ | $\gamma_2 \asymp d^{-\frac{4\alpha^2-4\alpha\beta}{2\alpha+2\beta-1}}$ (cf [80]) |

Table 13: $\kappa$ **and optimal** $\gamma_2$ **across the 4 phases for SGD**.

## J.2 Stochastic Gradient Descent

We will focus on SGD, although classical momentum can be handled entirely similarly. From [80, Sections I.1, I.2] batch has no effect on the compute-optimal frontier. We choose WLOG $B = 1$. We additionally know from [80, Prop. G.1, H.5] and Proposition G.5 that for $\gamma_2 Bt \lesssim d^{2\alpha}$ we have $\mathcal{K}(t,s) \asymp \gamma_2^2 B \min\{1, (\gamma_2 B(t-s))^{-2+1/(2\alpha)}\}$. It is hence clear that

$$\forall T \leq d^{2\alpha}, \quad \sup_{t \leq T} \int_0^t \mathcal{K}(t,s)\,\mathrm{d}s \asymp \gamma_2 \min\{1, (\gamma_2 BT)^{-1+1/(2\alpha)}\}.$$

Taking $\gamma_2 BT \asymp d^{2\alpha}$ we recover the stability for all time condition $\gamma_2 \lesssim d^{\max\{0,2\alpha-1\}}$. Additionally, if $\alpha > \frac{1}{2}$, we still get the stability condition $\gamma_2 \lesssim 1$ independent of $T$. However, for $\alpha < \frac{1}{2}$, we now obtain for $1 \lesssim \gamma_2 BT \lesssim d^{2\alpha}$ the condition $\gamma_2 \lesssim (\gamma_2 BT)^{-1/(2\alpha)+1} \gtrsim d^{2\alpha-1}$.

The question is now: for $\alpha < \frac{1}{2}$, can such a larger $\gamma_2$ improve the compute-optimal frontier? To that end, denote $T$ the time at which compute optimality is reached. Further introduce $\kappa$ such that $\gamma_2 BT = d^\kappa$. For $\alpha < \frac{1}{2}$, we hence obtain the condition $\gamma_2 \lesssim d^{-\kappa(-1+\frac{1}{2\alpha})}$.

We can now proceed to the same compute-optimal study as in Section E with the new $\kappa$ variable which gives results summarized in Table 13. Above the high-dimensional line, we obtain the same results as in Table 5. In phase I.b, since compute optimality is reached for $\gamma_2 BT$, we do not see improvement over Table 5. In phase I.c, we obtain $\gamma_2 \asymp d^{\frac{2\alpha(2\alpha-1)}{2\alpha+2\beta-1}} \gg d^{2\alpha-1}$ which yields faster compute-optimal curve. However in this phase, we do not have proofs about the kernel asymptotics which stops being power law. In fact it may be possible that a much larger learning rate can in fact be used. finally in phase IV, although the learning rate could be chosen larger, [80] showed that compute-optimality was reached by a smaller learning rate.

## J.3 DANA constant

The kernel is slightly more complicated in DANA constant and for simplicity we will use here a simplified form which becomes valid for $s \gtrsim \frac{\gamma_2}{\gamma_3}$ and $\frac{1+t}{1+s} \gtrsim \frac{1}{2}$ This will not affect the main results as this is the dominant term in the kernel norm. To see that, the reader can either notice that the upper-bound $\bar{\mathcal{K}}$ derived in Section H.10 is in fact tight, or directly integrate the estimates $\Phi_{11}, \Phi_{12}$ first on time and then on $\sigma$ for a complete proof.

$$\mathcal{K}(t,s) \asymp \mathcal{K}_1(t,s) + \mathcal{K}_2(t,s) \asymp \gamma_2^2 B(\gamma_2 B(t-s))^{-2+\frac{1}{2\alpha}} + \gamma_3((\gamma_2 B(t-s))^{-1+\frac{1}{2\alpha}}.$$

Again, denote for $T \geq 0$, $\gamma_2 BT = d^\kappa$ the compute-optimal time for some $\kappa > 0$.

As for SGD, we obtain the stability conditions

$$\begin{cases} \gamma_2 + \frac{\gamma_3}{\gamma_2 B}(\gamma_2 BT)^{1/(2\alpha)} \lesssim 1 & \text{if } 2\alpha > 1, \\ \gamma_2(\gamma_2 BT)^{-1+1/(2\alpha)} + \frac{\gamma_3}{\gamma_2 B}(\gamma_2 BT)^{1/(2\alpha)} \lesssim 1 & \text{if } 2\alpha < 1. \end{cases}$$

| | Optimal $\gamma_2$ | Optimal $\gamma_3$ |
|---|---|---|
| **Phase Ia** | $\gamma_2 \asymp 1$ | $\gamma_3 \asymp d^{-\frac{2\alpha}{4\alpha-1}}$ |
| **Phase IIa** | $\gamma_2 \asymp 1$ | $\gamma_3 \asymp d^{-\frac{2\alpha}{4\alpha-1}}$ |
| **Phase IIb** | $\gamma_2 \asymp 1$ | $\gamma_3 \asymp d^{-\frac{\alpha}{\beta(4\alpha-1)}}$ |
| **Phase IIIa** | $\gamma_2 \asymp 1$ | $\gamma_3 \asymp d^{-\frac{2\alpha}{4\alpha-1}}$ |
| **Phase IIIb** | $\gamma_2 \asymp 1$ | $\gamma_3 \asymp d^{-\frac{4\alpha}{(4\alpha-1)^2}}$ |

Table 14: **Optimal $\gamma_2, \gamma_3$ for DANA constant for $\alpha > \frac{1}{2}$. The results are valid up to a constant independent of the dimension**

Again, if we use the upper-bound $\gamma_2 BT = d^{2\alpha}$, we find the stability conditions $\gamma_2 \lesssim d^{\min\{0, 2\alpha-1\}}$ and $\frac{\gamma_3}{\gamma_2 B} \lesssim d$. However, these bounds are overly conservative, and we can use our study of DANA-constant to improve them by computing $\kappa$. For that, we use the write $\mathcal{P}(t) \asymp \mathcal{F}_0(t) + \mathcal{F}_{pp}(t) + \mathcal{F}_{ac}(t) + \frac{1}{\gamma_2 B}\mathcal{K}_{pp}(t,0)$ with $\mathcal{F}(t) = \mathcal{F}^{SGD}(\gamma_2 Bt \vee (\sqrt{\gamma_3 B}t)^2)$, $\mathcal{K}_{pp}(t,0) = \mathcal{K}_{pp}^{SGD}(\gamma_2 Bt \vee (\sqrt{\gamma_3 B}t)^2)$. We believe this can be made formal with an entirely similar proof as we did in Section B.3 by using our estimates of $\Phi_{11}(t,s), \Phi_{12}(t,s)$ although this is technical and not very enlightening. Again we will focus on $B = 1$ and more precisely on $\alpha > \frac{1}{2}$ which has slightly easier stability conditions. For $\alpha > \frac{1}{2}$, we necessarily have the bound $\gamma_2 \lesssim 1$. Therefore, we can easily compute $T$ as a function of $d, \gamma_3, B$ at compute optimality, and hence deduce $\frac{\gamma_3}{\gamma_2}$. Using this bound, we can also check that it is feasible which means $T \geq \frac{\gamma_2}{\gamma_3}$, which we used for the derivations, and hence find the results for phase Ia, II, III in Table 14.

Finally, one will note that $(\sqrt{\gamma_3 B}T)^2 \asymp \tau(t)^2$ the corresponding time of DANA-decaying around compute-optimality time $T$. This implies the following:

> The previously computed $\gamma_3$ for DANA-constant induces the same dynamics as DANA-decaying around the compute-optimal time $T$ after which DANA-constant starts diverging exponentially.

### J.4 DANA-decaying

Exactly as we did originally for DANA-decaying, we can try to construct a better learning rate than DANA-constant, by making sure that the value $\gamma_3$ wanted is attained at compute-optimality $\gamma_2 r = d^\kappa$. In other words, suppose that in some phase, DANA constant asks for $\gamma_3 = \gamma_2 B d^{-\kappa_3}$ and that compute optimality is attained at $\gamma_2 BT = d^\kappa$. Then DANA-decaying would use a schedule as $\gamma_3(t) = \gamma_2 B(\gamma_2 Bt)^{-\kappa_3/\kappa}$.

Computing these values from Table 14 **recovers exactly in all cases (at least above the high-dim)** $(1 + t)^{1/(2\alpha)}$. This is because this step-size is already optimal by ensuring that $\int_{s=0}^{T} \mathcal{K}_2(t,s)\,\mathrm{d}s \asymp \frac{\gamma_3(t)}{\gamma_2 B}(\gamma_2 BT)^{1/(2\alpha)} < 1$ is on the verge of diverging as $t \to \infty$. Note that this was also the criterion chosen for DANA-constant in the previous section at the point of compute optimality which explains the correspondence.

> Above the high-dimensional line, we can write the three equivalent characterizations of the DANA-decaying learning rate $\gamma_3(t) \stackrel{\text{def}}{=} \gamma_2 B(\gamma_2 B(1+t))^{-1/(2\alpha)}$:
>
> - the largest power law decay that satisfies the DANA-constant condition $\frac{\gamma_3(t)}{\gamma_2 B} \lesssim \frac{1}{d}$ for $t = d^{2\alpha}$ the time where the problem starts being strongly convex and the solutions stop behaving as power law and decay-exponentially,

- the largest power law such that the stochastic noise induced by momentum $\int_0^t \mathcal{K}_2(t,s)\,\mathrm{d}s \asymp \frac{\gamma_3(t)}{\gamma_2 B}(\gamma_2 Bt)^{1/(2\alpha)}$ remains bounded for all time $t \geq 0$,

- the power law which recovers $\gamma_3(T) = \gamma_3$ the constant DANA-constant learning rate computed in section Section J.3 for which the kernel norm remains bounded up to the compute-optimal time $T$.

## K   Power-Law Random Features Experiments & Numerical Simulations

In this section, we describe the PLRF experiments, measurements of the empirical scaling law exponents, and the numerical simulations of the ODEs shown in Figure 8, Figures 25-33 and Figures 34-42.

### K.1   Power-Law Random Features Experiment Details

For each $(\alpha, \beta)$ pair, we run the stochastic algorithms for SGD, DANA-constant, and DANA-decaying on PLRF with parameters

$$d \in [200, 300, 400, 600, 800, 1200, 1600, 2400, 3200, 4800, 6400, 9600, 12800].$$

and set $v = 4 \times d$. We selected 84 pairs of $(\alpha, \beta)$'s that provide a good representation of all the different phases spanning $\alpha \in [0.2, 2]$ and $\beta \in [-0.15, 1.4]$. We use batch size $= 1$ for all PLRF experiments and compute the mean population risk across a set of random seeds. We use 10 random seeds for most experiments but use 100 random seeds when $\beta \leq 0.3$ due to the increased gradient noise when $\beta$ is small.

We calculate flops[14] as the number of training steps $\times d$. We train all models for $1e12$ flops or convergence, whichever occurs sooner.

The loss curves in Figures 25-33 and Figures 34-42 show nearly perfect agreement with the numerical simulations of the simplified ODEs (23) across all three algorithms and all values of $(\alpha, \beta)$ and $d$. In particular, this numerical agreement validates that the simplification in the ODEs between (6) and (7) has a trivial numerical effect.

We define $\mathrm{Tr}(D) \stackrel{\text{def}}{=} \sum_{i=1}^{d} j^{-2\alpha}$.

### K.2   Measuring the Empirical Scaling Law Exponents: Chinchilla Approach 1

To measure the empirical scaling law exponents for the loss and parameter count, we follow Chinchilla Approach 1 [50]. We note that for SGD on PLRF, similar experiments in [80] found close agreement for exponent measurements between Chinchilla Approach 1 and Approach 2. We do not expect perfect agreement between theoretical predictions for the scaling exponents due to noise in the loss curves, discretization in the parameter counts, and finite-size effects noted in Approach 0 in [80] that showed the 'instantaneous' power-law exponents vary as a function of flops well beyond practical scales for experimentation.

We follow Chinchilla Approach 1 as described in Section 3.1 of [50]. First, we choose a flops window $[\mathfrak{f}_{\min}, \mathfrak{f}_{\max}]$ and construct $\mathfrak{f}_j$'s using a geometric spacing between $\mathfrak{f}_1 = \mathfrak{f}_{\min}$ and $\mathfrak{f}_n = \mathfrak{f}_{\max}$. For each flops slice $\mathfrak{f}_j$, we find the minimum loss across all $d$. We denote this minimum value of the loss, $\mathscr{P}^\star(\mathfrak{f}_j)$, and its associated parameter count, $d^\star(\mathfrak{f}_j)$. This creates an 'envelope' of optimal losses vs flops as shown in Figures 25-33 or a 'staircase' of optimal parameter count vs flops as shown in Figures 34-42.

The compute-optimal loss exponent is found by plotting

$$\{(\mathfrak{f}_j, \mathscr{P}^\star(\mathfrak{f}_j))\}_{1 \leq j \leq n}, \tag{130}$$

and fitting a power law curve of the form $\mathscr{P}^\star(\mathfrak{f}) = c \times \mathfrak{f}^{-\hat{\eta}}$. For the 84 different pairs of $(\alpha, \beta)$, we plot the fitted power law exponents and list the predicted theoretical loss exponents (see Table 5 (SGD/SGD-M), Table 6 (DANA-constant), and Table 7 (DANA-decaying)) in the legends in Figures 25, 26, 27 for SGD, Figures 28, 29, 30 for DANA-constant, and Figures 31, 32, 33 for DANA-decaying. The (absolute) maximum difference between theory vs. empirical is 0.09. Figure 8 (middle) used the loss exponents from Figures 25-33.

The compute-optimal parameter exponent is found by plotting,

$$\{(\mathfrak{f}_j, d^\star(\mathfrak{f}_j))\}_{1 \leq j \leq n}. \tag{131}$$

---

[14]Note that for consistency with the theoretical setup, for the PLRF experiments we omit the 6 in the $6 \times$ num parameters $\times$ tokens heuristic [50, 54] often used for calculating flops. We include this 6 in the LSTM experiments for accuracy and consistency with the empirical scaling laws literature.

and fitting the function $d^\star = c \times \mathfrak{f}^b$ where $c, b$ are constants. The empirical parameter exponent measurement is given by the exponent $b$. For the 84 different pairs of $(\alpha, \beta)$, we plot the fitted power law exponents and list the predicted theoretical parameter exponents in the legends in Figures 34, 35, 36 for SGD, Figures 37, 38, 39 for DANA-constant, and Figures 40, 41, 42. The predicted theoretical parameter exponents can be found in Table 5 (SGD/SGD-M), Table 6 (DANA-constant), and Table 7 (DANA-decaying). The absolute maximum difference between theory and Approach 1 is 0.132. Figure 8 (right) used the parameter exponents from Figures 34-42.

It is important to note that the fit of the exponents is sensitive to the choice of the flops window; changing the window can have a dramatic effect on the predictions. For Approach 1, we have a discrete set of parameter counts $d$ that approximate the 'true' compute-optimal frontier that would result from an arbitrarily dense sweep over parameter counts. We therefore want to fit the power laws for Approach 1 within the range of flops where the parameter counts in our experiments act as a good approximation of the compute-optimal frontier. More specifically, as a heuristic, we want to fit Approach 1 within the flops window whose minimum is the flops where the two smallest models crossover, and whose maximum is the flops where the two largest models crossover. In particular, for larger values of alpha, the loss curves from the larger values of $d$ do not intersect because the experiment could not run long enough, so we want the flops window maximum to be the crossover between the two largest models that did reach a crossover.

As a first pass, we used this heuristic to determine the flops window for each algorithm and $(\alpha, \beta)$ pair using smoothed loss curves to locate the crossover points between model sizes. However, due to noise in the loss curves, it is nontrivial to systematically distinguish which crossovers in the noisy curves are due to noise and which represent the location of the 'true' crossover that would result if averaging over an arbitrarily large number of random seeds. We therefore manually adjusted the flops windows using visual inspection to adhere to the intent of this heuristic.

### K.3 Computing the deterministic equivalent for $\hat{K}$

In order to derive a fully deterministic curve for the loss, we need get a precise estimate for the deterministic equivalent of $\hat{K} = D^{1/2}WW^T D^{1/2}$ where $D = \mathrm{Diag}(j^{-2\alpha} : 1 \le j \le v\}$. In particular, we need to solve the fixed point equation in (33),

$$m(z) = \frac{1}{1 + \frac{1}{d}\sum_{j=1}^{v} \frac{j^{-2\alpha}}{j^{-2\alpha}m(z)-z}} \quad \text{where} \quad \mathcal{R}(z) = \mathrm{Diag}\left(\frac{1}{j^{-2\alpha}m(z)-z} : 1 \le j \le v\right). \tag{132}$$

In order to implement and solve such a fixed point equation, given that $\|\hat{K}\|_{\mathrm{op}} \approx 1$, we uniformly discretize the $z$'s from $\epsilon \times j^{-2d}$ to $1 + \epsilon$ for some small $\epsilon$ and add a small imaginary component. For each of these $z$'s values, we solve the fixed point equation in (33) with Newton's method constrained so that the Newton update always remains in the upper half plane.

### K.4 Implementation of the ODE

To implement the ODEs in (23), we use *implicit Euler* with step $h$ and use the deterministic equivalent as the initialization. While other ODEs can be used, implicit Euler is known to work well on problems whose solution is exponentially decreasing, but ill-conditioned – exactly our setting. To improve the speed of implicit Euler, we perform a time change mapping $t \mapsto e^t$. This allows for log-spaced time and an exponential speed up in solving the ODEs. We note that one does need to solve $d$ ODEs simultaneously, but the ODEs form a nearly decoupled system of 3 ODEs as the coupling only occurs through the forcing function ($\mathscr{P}(t)$). Numerical experiments in this section (Sec. K) and Fig. 8 use a step length of $h = 10^{-3}$ (after changing into log spacing). All other numerical experiments in the paper use a step length of $h = 10^{-2}$.

# L   LSTM Language Model Experiments

In this section we present experiments for SGD and DANA-decaying on LSTMs trained on text data. We constructed a language model setup using a cross-entropy loss on next-token prediction on the C4 text dataset [84]. We use the LSTM [49] architecture from [53] with standard parameterization.

To sweep model sizes, we co-scale the embedding and hidden dimensions while keeping the depth (number of layers), sequence length, batch size and vocabulary size fixed. The embedding sizes used are $[16, 24, 32, 48, 64, 96, 128, 192, 256, 512, 1024, 2048, 4096]$ and the hidden dim is set to $2 \times$ emb dim. All models use depth = 2 layers, sequence length = 20 and batch size = 32.

## L.1   LSTM Results: DANA-decaying across $\kappa_3$

In this section, we train models with SGD with learning rate $=$ 0.5 and DANA-decaying with $\gamma_2 = 0.5$ and $\gamma_3(t) = \gamma_2 \times (1 + t)^{-\kappa_3}$ where we sweep $\kappa_3 \in [0.0, \dots, 1.4]$.

We first consider a single model size for the LSTMs (embedding dim = 128). In Figure 19, we repeat Figure 3a on the left and Figure 2a on the right to show side-by-side that the behavior of DANA-decaying across $\kappa_3$ is qualitatively similar on PLRF (left) and LSTM (right). On PLRF, we see that DANA-decaying diverges when $\kappa_3 < 1/2\alpha$. On LSTMs, we see that DANA-decaying similarly diverges for $\kappa_3 < 0.6$. Moreover, we see that for moderate values of $\kappa_3$ DANA-decaying outperforms SGD in both model settings, and that as $\kappa_3$ approaches 1.0, DANA-decaying smoothly deteriorates back to SGD.

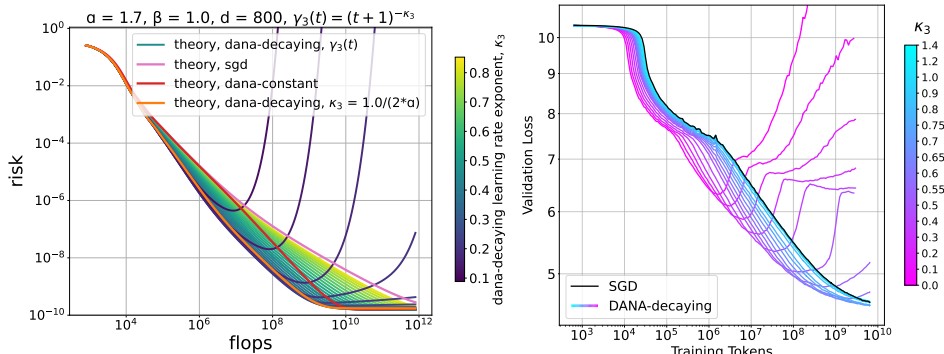

Figure 19: **DANA-decaying shows similar stability and divergence behavior across $\kappa_3$ on PLRF and LSTMs.** *(left)* We repeat Figure 3a on PLRF where $\gamma_3(t) \asymp (1 + t)^{-\kappa_3}$. This shows DANA-decaying diverges when $\kappa_3 < 1/2\alpha$ and degrades to SGD as $\kappa_3$ goes towards 1.0. *(right)* We repeat Figure 2a on LSTMs with embedding size = 128 showing DANA-decaying diverging when $\kappa_3 < 0.6$ and degrading to SGD when $\kappa_3$ exceeds 1.0.

We next repeat this experiment across model sizes, with embedding sizes ranging from 16 to 4,096. In Figure 20 we see that the behavior of DANA-decaying is qualitatively similar across model sizes: we continue to see divergence on small $\kappa_3$ values, outperforming SGD on moderate $\kappa_3$ values, and degrading back to SGD on $\kappa_3$ values above 1.0.

One interesting question is whether the optimal value of $\kappa_3$ is consistent across model sizes. To investigate this, we plot the final loss versus $\kappa_3$ in Figure 21. We see that the optimal $\kappa_3$ is fairly consistent across model sizes, with the optimal value of $\kappa_3$ falling within 0.55 to 0.75 on all model sizes with a trend towards 0.75 for the large models. We note that the largest models may slightly overestimate the optimal $\kappa_3$ if they did not run far enough to reach divergence. Using the PLRF as an analogue where $\kappa_3 = \frac{1}{2\alpha}$, this optimal $\kappa_3$ near 0.75 would correspond to $\alpha = 0.67$, but we note that the precise meaning of $\alpha$ is not clear for LSTMs. We leave this question about how to measure and interpret $\alpha$ on real-world problems for future work.

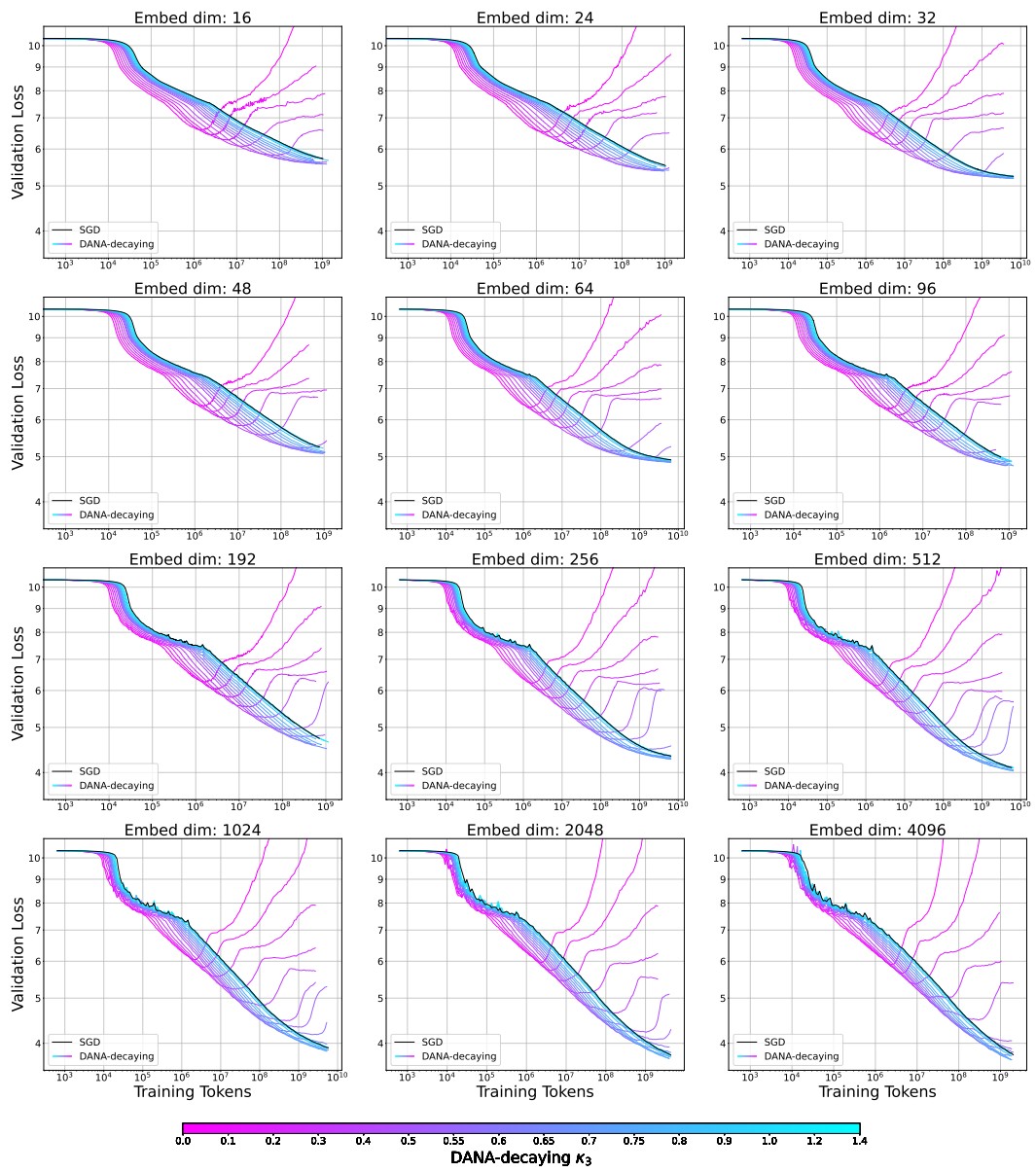

Figure 20: **Sweeps of $\kappa_3$ for DANA decaying show qualitatively similar behavior across model sizes.** Small values of $\kappa_3$ diverge, moderate values of $\kappa_3$ outperform SGD, and large values of $\kappa_3$ trend back towards SGD. Each panel shows one model size with SGD in black and DANA-decaying in bright colors indicating the value of $\kappa_3$.

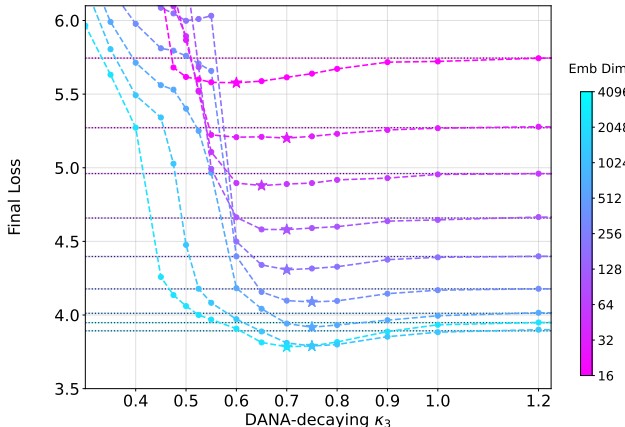

Figure 21: **Final loss versus DANA-decaying $\kappa_3$ showing the $\kappa_3$ which optimizes final loss is fairly consistent across model sizes.** The optimal value of $\kappa_3$ for the final loss is between $0.55$ to $0.75$ on all models with a trend towards $0.75$ for the largest model sizes. Color indicates model size. Black dotted lines indicate the final SGD loss for the model size indicated by the color of the highlight.

## L.2 LSTM Results: Loss Exponents

We next measure the LSTM loss exponents for SGD and for each individual value of $\kappa_3$ for DANA-decaying, by plotting the loss curves across model sizes and considering the compute-optimal frontier defined by the lower envelope of these curves. We observe that the compute-optimal frontier appears to follow a power-law trend for an intermediate set of models, but the power law breaks on the largest model sizes similar to what was observed in Figure 7 of [54] on a similar LSTM scaling setup.

We perform a version of Chinchilla Approach 1 [50] using an intermediate set of model sizes ranging from embedding size 16 to 192. We use an abbreviated version of this method where we fit power laws through the crossover points in the loss curves between adjacent model sizes. Due to the smoothness in the loss curves, the locations of the crossover points between model sizes can be precisely determined. The power law fits have very high $R^2$ values, exceeding 0.99 on all settings except the most unstable $\kappa_3 = 0.0$ which has $R^2 = 0.984$.

In Figure 22, we show the loss curves and compute-optimal power law fits for each algorithm and value of $\kappa_3$, and report the loss exponents and $R^2$ values in the legends. Finally, in Figure 23a we plot the loss exponents as a function of $\kappa_3$ and show the SGD loss exponent as a baseline. The magnitude of the DANA-decaying loss exponent is maximized when $\kappa_3 = 0.7$ and exceeds the SGD loss exponent magnitude. We note that the measurement of $\kappa_3 = 0.7$ that maximizes the loss exponent magnitude is very close to the $\kappa_3$ near $0.75$ that optimizes the final loss, which is interesting because the loss exponent measurement is determined using earlier phases of training whereas the final loss values naturally describe the behavior from late in training. While the numerical differences between the SGD and DANA exponents are small, the high $R^2$ values and the smoothness of the DANA-decaying loss exponents as a function of $\kappa_3$ indicates that these measurements might be robust and imply a true improvement in the loss exponent over SGD.

More notably, the DANA-decaying loss exponents traverse exactly the regimes seen in PLRF: diverging for small values of $\kappa_3$, outscaling SGD for intermediate values of $\kappa_3$, and reverting back to SGD-like scaling for $\kappa_3 \geq 1.0$. We note that $\kappa_3 = 1.0$ corresponds to Schedule-Free SGD and shows almost exactly the same loss exponent as SGD for LSTMs.

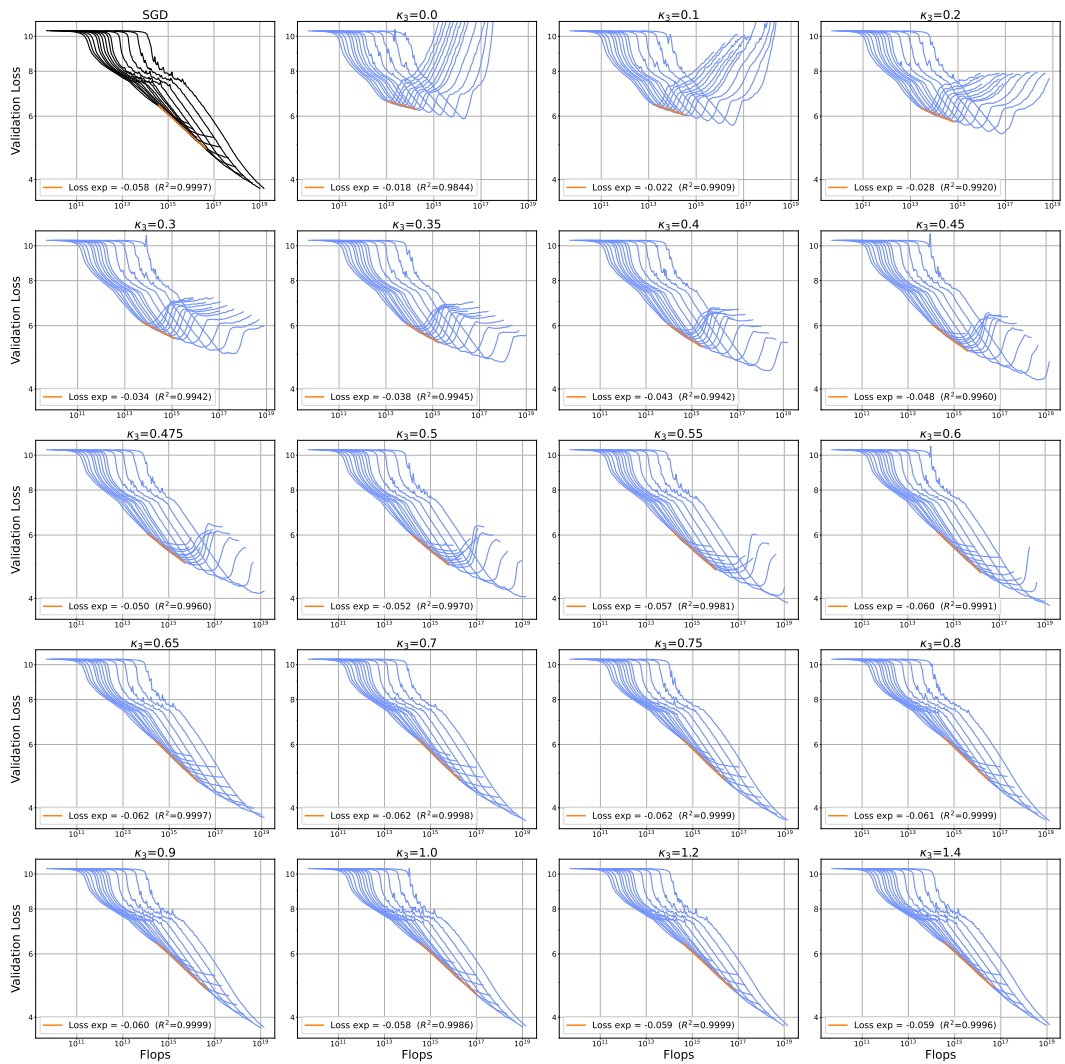

Figure 22: **Loss exponents for SGD and DANA-decaying across** $\kappa_3$**.** Loss curves across model sizes (black or light blue) and power law fits (orange) through the compute-optimal frontiers showing the loss exponent measurements (with $R^2$ values) for SGD and DANA-decaying. The first panel shows SGD and remaining panels each represent one value of $\kappa_3$ for DANA-decaying.

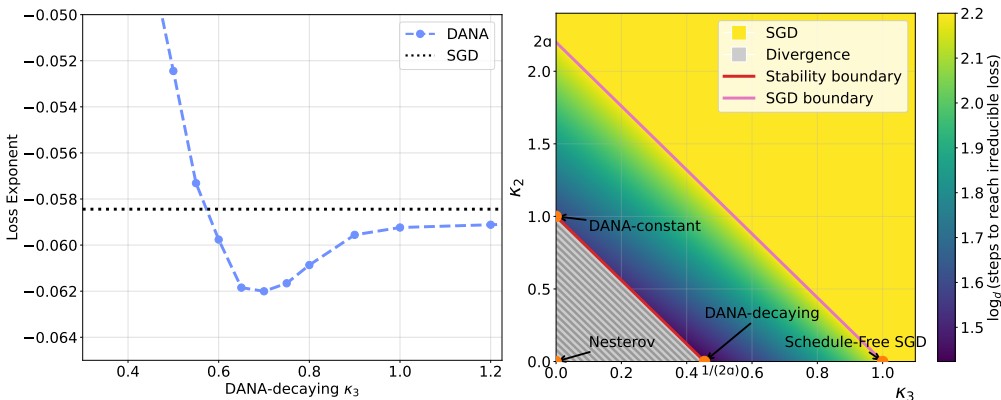

Figure 23: **DANA-decaying loss exponents show improvement over SGD, and traverse the regimes seen in PLRF.** *(left)* Loss exponent versus $\kappa_3$ for DANA-decaying. The loss exponent for SGD is shown with a black dotted line. The loss exponents for DANA-decaying have higher magnitude indicating an improvement in the loss exponent for $0.6 \leq \kappa_3 \leq 0.9$. *(right)* We repeat Figure 6 showing the regimes for PLRF as a function of $\kappa_3$. We note that the LSTM loss exponents (left) traverse the same regimes as the $x$-axis of the right figure: divergence for small values of $\kappa_3$, outscaling SGD for intermediate values of $\kappa_3$, and reverting back to SGD-like scaling for $\kappa_3 \geq 1.0$. Note $\kappa_3 = 1.0$ corresponds to Schedule-Free SGD, and shows almost exactly the same loss exponent as SGD for LSTMs.

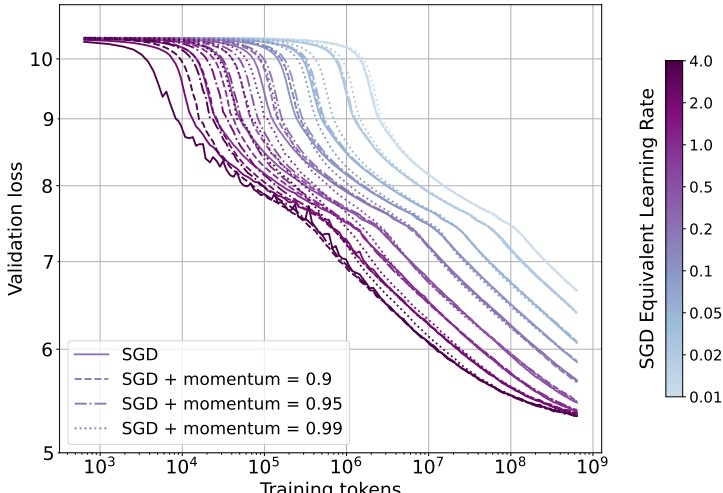

Figure 24: **The equivalence in risk dynamics between SGD and SGD-M holds approximately in LSTMs.** We sweep different values of momentum (different line styles) and different effective learning rates where effective learning rate = learning rate / (1 - momentum). Color corresponds to the effective SGD learning rate. Loss curves of the same color have similar dynamics showing that the equivalence in PLRF dynamics extends fairly closely to the LSTM setting.

## L.3 Equivalent risk dynamics for SGD-M and SGD

As noted in Section 5 and Remark G.2, there is an equivalence in risk dynamics on PLRF when $\gamma_2^{\text{SGD}} = \gamma_2^{\text{SGD-M}} + \gamma_3^{\text{SGD-M}}/\delta$.

In Figure 24, we show this equivalence holds closely in the LSTM setting especially after the early phase in training. We perform a two-dimensional sweep across learning rates and momentum values. We train SGD (corresponding to $m = 1.0$) and SGD-M with three different momentum values of $m \in [0.9, 0.95, 0.99]$. Following the equivalence, we sweep the SGD "effective learning rate" denoted by $\eta$ and set the learning rate to $\eta/(1-m)$ for each value of $m$. We sweep $\eta \in [0.01, 0.02, 0.05, 0.1, 0.2, 0.5, 1.0, 2.0, 4.0]$.

In Figure 24, we see that all values of momentum have similar risk dynamics for the same effective learning rate (same color), particularly after the early phase of training. While we do not measure the loss exponent for SGD-M directly, this equivalence suggests that SGD-M should have nearly identical scaling behavior to SGD and result in the same loss exponent.

## L.4 Experiment Details

**Initialization.** We use standard parameterization to initialize the LSTM parameters. All matrix parameters use Gaussian initialization with zero mean. The embedding parameters are initialized with standard deviation = 0.1 and the hidden (LSTM cell kernel and recurrent) parameters and readout parameters are initialized with standard deviation $1/\sqrt{\text{fan-in}}$. Bias parameters are initialized to zero.

**Dataset.** All models are trained on the C4 language dataset [84] encoded with the T5 SentencePiece [57] tokenizer, with an additional beginning-of-sequence (BOS) token, resulting in the vocabulary size of $V = 32,001$ (32,000 original vocabulary + 1 BOS). The effective vocabulary dimension in experiments is 32,101 due to 100 unused tokens. Both training inputs and evaluation inputs are padded rather than sequence-packed. We use the 'train' split for training and the 'validation' split for evaluation from the TFDS [1] version of the dataset.

We train in the single epoch regime: the C4 dataset contains 156 billion tokens, but our models never train past the first few billion tokens and we never repeat training examples.

**Implementation details.** Our LSTM implementation started from a fork of the code released with [53], with changes made for performance, initialization, and switching the dataset. The LSTM

implementation uses JAX [32] with Optax optimizers [32]. We use a modified version of the data pipeline from NanoDO [65].

**Hyperparameters.** We first swept the SGD learning rate and selected a stable learning rate of $0.5$ which we use for all SGD experiments except the learning rate sweeps in Fig. 24.

For DANA-decaying, we set $\gamma_2 = 0.5$ to correspond to the SGD learning rate, and $\gamma_3(t) = \gamma_2 \times (1 + t)^{-\kappa_3}$. We then sweep over $\kappa_3$; the minimum $\kappa_3$ where DANA-decaying converges appears optimal (Fig. 2c & 20). Since $\delta$ just needs to be large, we set $\delta = 8$.

**Flops calculation.** We compute flops as $6 \times$ non-embedding parameters $\times$ training tokens using the '6ND' heuristic [50, 54]. We follow the common practice of counting only the non-embedding parameters as this has been shown in [50, 54] to induce better power law fits.

## M   Additional Information on the Experimental Set-Ups for Figures

In this section, we provide additional information regarding the experimental set-ups used to generate some of the figures. For most figures, we have included these in the captions. Due to space limitations, some figure experimental set-ups are included here.

**Figure 1a.**   Experimental setup: $d = 500$, $v = 2500$, batch size = 1, $10^7$ iterations, with learning rates $\gamma_2 = 0.5/\text{Tr}(D)$ for all methods, $\gamma_3 = 0.1/(\text{Tr}(D) \times d)$ for DANA-constant, and $\gamma_3(t) = 0.1(1 + t)^{-1/(2\alpha)}$ for DANA-decay.

**Figure 1b.**   Numerical setup: SGD (blue curves) learning rate $\gamma_2 = 0.5/\text{Tr}(D)$, DANA-constant (green) has $\gamma_1(t) = 1$, $\gamma_2(t) = 0.5/\text{Tr}(D)$, $\gamma_3(t) = 0.1/d$, $\Delta(t) = \delta/(1 + t)$ where $\delta = \max\{2 - 1/\alpha, (2\alpha + 2\beta - 1)/\alpha\} + 1$; DANA-decaying (orange) same $\gamma_1, \gamma_2$, and $\Delta$ as DANA-constant, $\gamma_3(t) = 0.1/(1 + t)^{1/(2\alpha)}$; Schedule-free SGD (red) $\tilde{\beta} = 0.9$ and $\tilde{\gamma} = 0.5/\text{Tr}(D)$ in [33]. Algorithms run using the ODEs (43) with hyperparameters given in Table 3 for Schedule-Free SGD; $10^9$ iterations of algorithm, $d = \{100 \times 2^i\}$, $i = 1, \ldots, 10$ and $v = 10 \times d$. Schedule-free SGD (red) scales very closely with SGD (blue) for this $(\alpha, \beta)$. We see both DANA-decay and DANA-constant accelerate.

**Figure 1c.**   PLRF setup: number of iterations for all algorithms is $10^7$, $\alpha = 1.2, \beta = 0.7$, batch size 1 with $d = \{250, 500, 1000, 2000\}$ and $v = d \times 2$. Adam [56] algorithm was run from Optax [32] where the default parameters (i.e., $\beta_1 = 0.9$, $\beta_2 = 0.999$, $\varepsilon = 10^{-8}$) were applied. For the Adam learning rate, a cosine-decay schedule, also from Optax, was used. Initial value for the cosine learning rate is $0.001$. For each $d$-value, multiple runs of adam with the decay steps in the cosine-decay learning rate given by $2 \times 1000 \times m$, $m = 0, 1, \ldots$ until decay steps is greater than total iterations, $10^7$. For each $d$, (solid green) lower envelope of the Adam runs; (faded green) one single run of Adam with a fixed decay step. SGD (solid orange) and DANA-decaying (sold blue) plotted using ODEs. DANA-decaying hyperparameters: $\delta = \max\{2 + (2\beta - 1)/\alpha), 2 - 1/\alpha\} + 1$ with $\Delta(t) = \delta/(1 + t)$; $\gamma_3(t) = 0.1/\text{Tr}(D) \times (1 + t)^{-1/(2\alpha)}$; $\gamma_2(t) = 0.3/\text{Tr}(D)$; $\gamma_1(t) = 1$. SGD hyperparameters: $\gamma_2(t) = 0.3/\text{Tr}(D)$.

**Figure 3a and Figure 3b.**   Optimal learning rate schedule, varying $\kappa_3$ and $\kappa_2$ in learning rate $\gamma_3(t)$. Numerical set-up: $d = 800$ and $v = 5 \times d$; batch size is 1, number of iterations is $10^9$. SGD: learning rate is $\gamma_2 \equiv 0.5/\text{Tr}(D)$ where $\text{Tr}(D) = \sum_{j=1}^{v} j^{-2\alpha}$; DANA-constant: $\gamma_2$ same as SGD, $\gamma_3(t) = 0.1/(\text{Tr}(D)) \times d^{-\kappa_3}$ where $\kappa_3$, if not specified is 1, $\Delta(t) = \delta/(1 + t)$ where $\delta = \max\{2 + (2\beta - 1)/\alpha, 2 - 1/\alpha\} + 1$; DANA-decaying: $\gamma_2$ same as SGD, $\gamma_3(t) = 0.1(1 + t)^{-\kappa_3} \times 1/\text{Tr}(D)$ where $\kappa_3$, if not specified, is $1/(2\alpha)$, $\Delta(t)$ same as DANA-constant. Curves generated by using the ODEs in (7) with the deterministic equivalent for $\hat{K}$; In Fig. 3a, if $\kappa_3 < 1/(2\alpha)$ in $\gamma_3$, DANA-decaying will eventually diverge, but until it does it exactly follows the curve for DANA-decaying with $\kappa_3 = 1/(2\alpha)$. Moreover when $\kappa_3 = 1/(2\alpha)$ in Fig. 3a, DANA-decaying does not diverge and it appears optimal. When $\kappa_3 = 1$, i.e., momentum is a pure average, the loss curve is similar to SGD and does not accelerate. Similarly, in Fig. 3b, for DANA-constant with $\gamma_3(t) \asymp d^{-\kappa_2}$, $\kappa_2 = 1$ is the only power in $d$ for which the algorithm does not diverge, however the algorithm is slower. For $\kappa_2 < 1$, initially the curves decrease before diverging and, up until divergence, follow the trajectory of DANA-decaying with $\gamma_3(t) \asymp (1 + t)^{-1/(2\alpha)}$. This justifies the learning rate schedule $\gamma_3(t) \asymp t^{-1/(2\alpha)}$ in DANA-decaying.

**Figure 3c.**   Numerical set-up: 100 randomly generated $\hat{K} = DWW^T D$ and the $\rho_j$'s computed; for empirical density $\mu_{\mathcal{F}}$, 500 bins equal spaced on log scale from $10^{-8}$ to 1 and counted the number of $\lambda_j$ that fall into each bin weighted by $\rho_j$'s and then averaged over the 500; For deterministic equivalent $\mu_{\mathcal{F}}$, solved fixed pointed equation (33) using Newton Method on a grid of $x$-values.

**Figure 6.**   Suppose $\kappa_1 = 0$ and we are above the high-dimensional line, $2\alpha > 1$. Using Theorem 4.1, you can find the number of iterations needed to reach the optimum, $\mathcal{F}_0$, when $2\alpha > 1$; it is independent of which phase (Phase Ia, II, III). In particular, we have that

$$\vartheta(t) \asymp \max\{t, d^{-\kappa_2} t^{-\kappa_3 + 2}\}.$$

For $2\alpha > 1$ and $B = 1$, a quick calculation shows that

$$\text{time to reach irreducible loss}, t \asymp d^\eta \quad \text{where} \quad \eta = \begin{cases} \min\left\{\frac{2\alpha + \kappa_2}{2 - \kappa_3}, 2\alpha\right\}, & \text{if } \frac{2\alpha + \kappa_2}{2 - \kappa_3} > 0 \\ 2\alpha, & \text{else.} \end{cases}$$

When $t \asymp d^{2\alpha}$, this is the same amount of time that SGD requires to reach the irreducible loss level. Equality holds in the minimum exactly at the (magenta) line, $\kappa_2 = 2\alpha(1 - \kappa_3)$.

In terms of divergence, we can look at the stability condition (stability) in Theorem 4.1. From this, we have divergence if $\kappa_2 \leq -2\alpha \times \kappa_3 + 1$. This is precisely below the (red) line. We plotted the values of $\eta$ for $\alpha = 1.1$.

**Figure 7.** Numerical set-up: $d = 100 \times 2^i$, $i = 1, \ldots, 10$ for Phase I, IIa, IIIa, $d = 100 \times 2^i$, $i = 10, \ldots, 15$ included for Phase IIb, IIIb; $\delta \geq \max\{2 + (2 + \beta - 1)/\alpha, 2 - 1/\alpha\}$ with $v = 5 \times d$; SGD: learning rate is $\gamma_2 \equiv 0.5/\text{Tr}(D)$; DANA-constant: $\gamma_2$ same as SGD, $\gamma_3(t) = 0.15/(\text{Tr}(D)) \times d^{-1}$, $\Delta(t) = \delta/(1+t)$ where $\delta = \max\{2 + (2\beta - 1)/\alpha, 2 - 1/\alpha\} + 1$; DANA-decaying: $\gamma_2$ same as SGD, $\gamma_3(t) = 0.25(1 + t)^{-1}$, $\Delta(t)$ same as DANA-constant; colored lines computed using deterministic ODEs (7) over different $d$ and black lines the predicted compute-optimal curves from Table 5,6,7. Our predictions match the scaling laws of the deterministic ODEs and show that DANA-decaying yields better scaling laws. **Phase diagram, (bottom, right).** Green region: both DANA-constant and DANA-decaying accelerate; red dashed region: DANA-constant doesn't accelerate but DANA-decaying does; red region: neither DANA-constant nor DANA-decaying accelerate.

**Figure 8.** *(left)* Simplified ODEs (43) and exact ODEs (22) match the stochastic algorithms across multiple $d$-values. Prediction for the loss exponents match empirical (Approach 1 used from [50]). *(middle)* Plotted is the loss exponent $\eta$, $\mathcal{P}(\mathfrak{f}/d^\star; d^\star) = \mathfrak{f}^\eta$ against the our predicted $\eta$'s given in theory (see Table 6, 5, 7) by fixing $\beta = 0.7$ and going through different $\alpha$ values; as $\alpha \uparrow$ go through Phase Ic $\to$ IVa $\to$ IVb $\to$ III $\to$ II. Solid dots computed from multiple runs of the stochastic algorithm using Approach 1 from [50] (see Section K for details). Predictions match well estimated values and show that as SGD, DANA-constant, and DANA-decaying all coincide when in IV. Exponent of DANA-constant and DANA-decay are significantly larger than SGD in III and II. In Ic, there is a small discrepancy with theory versus estimated due to dimensonality, $d$, effects (see [80] for more details). *(right)* Same plot for params exponents (top), $\xi$ on $d^\star = \mathfrak{f}^\xi$ and data exponents (bottom), $\zeta$, i.e., exponent at compute-optimum for number of samples; Fixed $\alpha = 1.0$ and varying $\beta$.

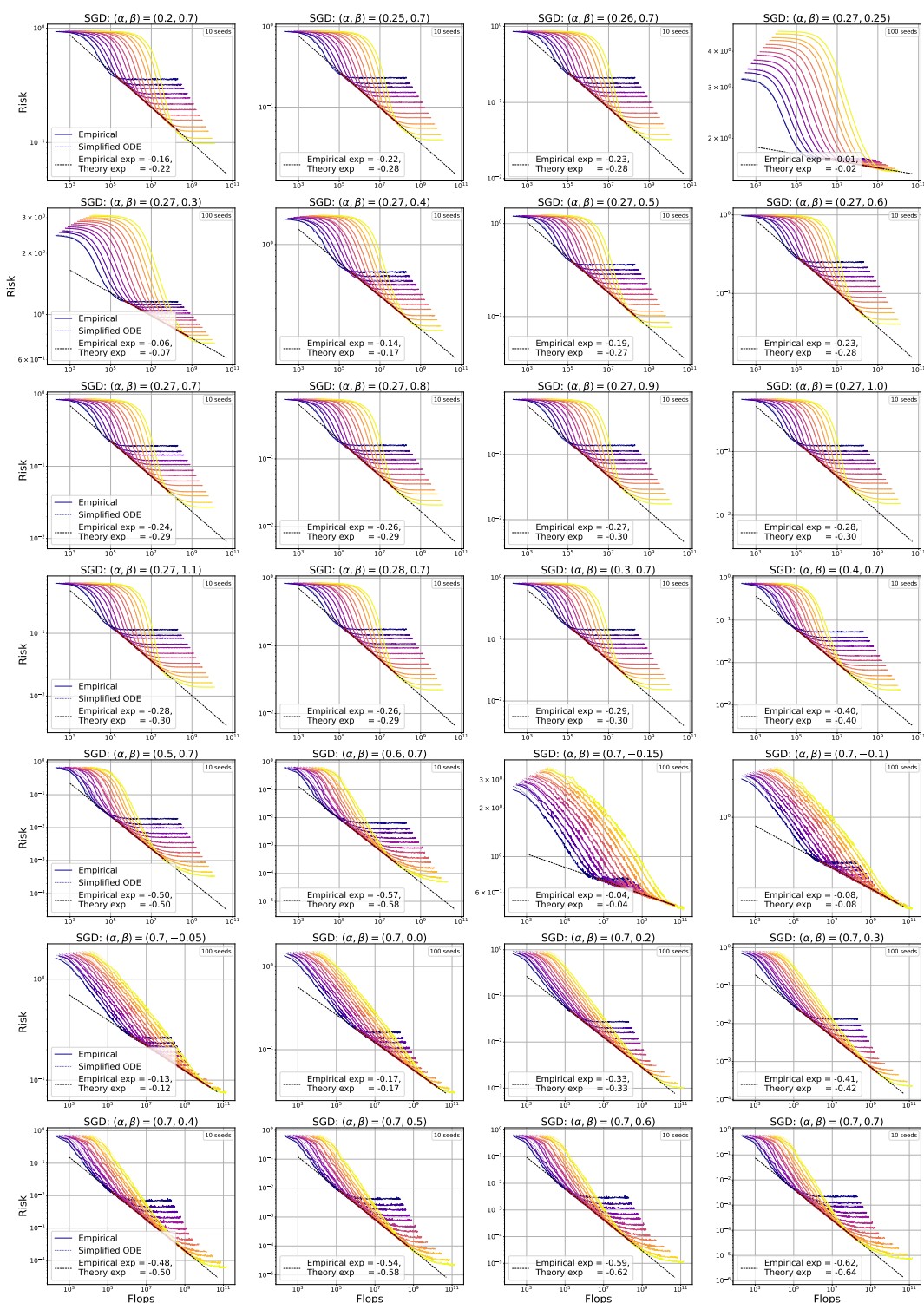

Figure 25: **SGD loss curves and compute-optimal loss vs. theory on PLRF.** For various $(\alpha, \beta)$, we plot the mean population risk for stochastic algorithm runs over 10-100 seeds (see individual figures) for SGD (solid lines) with $\gamma_2 = 0.375/\text{Tr}(D)$. Colors indicate dimensionality $d$ ranging from 200 to 12,800. The stochastic runs of SGD match the solutions of the simplified ODEs in (43) (dotted lines) nearly perfectly. The empirical compute-optimal power-law (dashed black line) is generated using Chinchilla Approach 1 [50] where the solid red highlighted section shows the region fit. The empirical compute-optimal loss exponents nearly match theory for all $(\alpha, \beta)$ tested, with absolute maximum difference of 0.08.

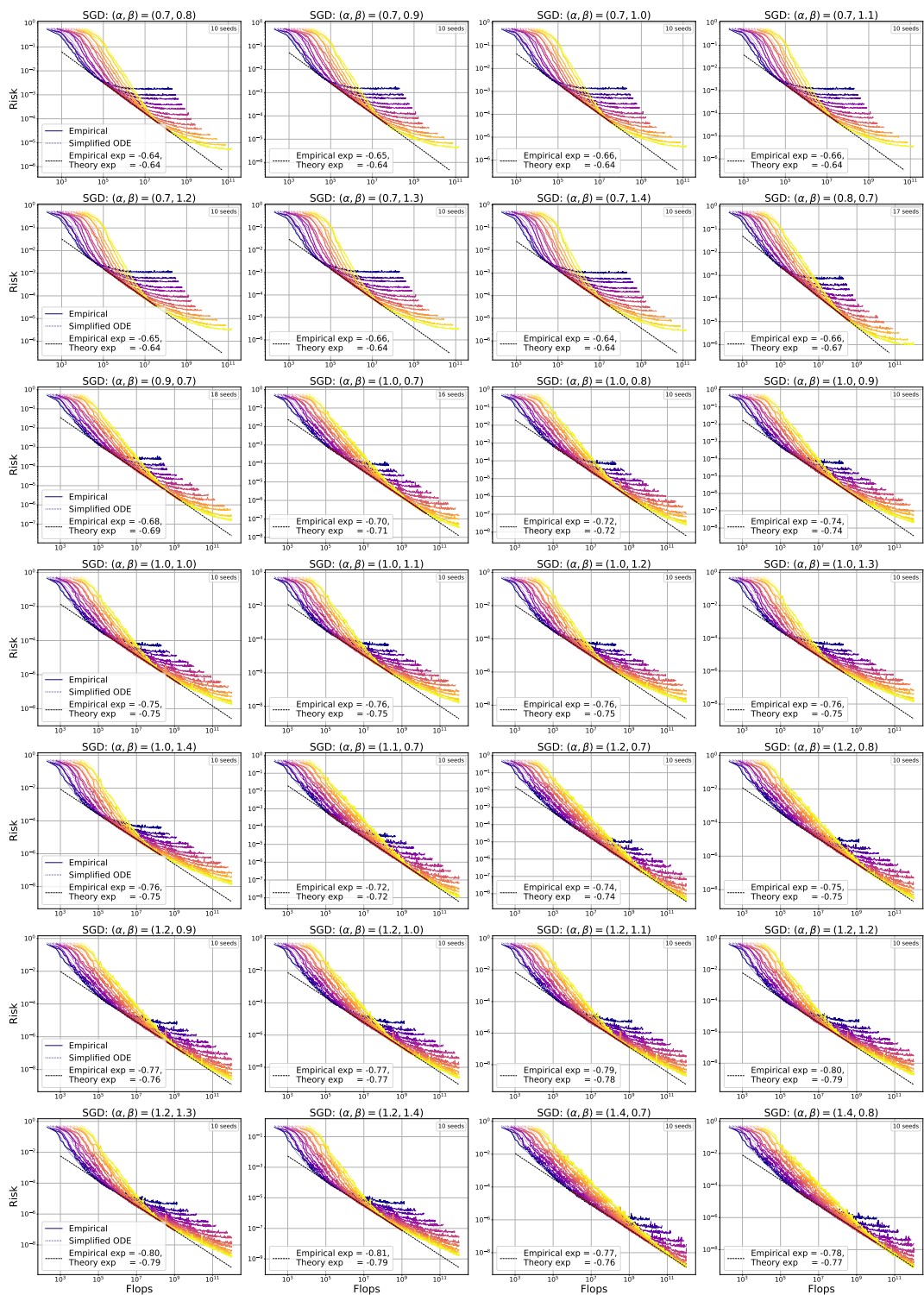

Figure 26: **SGD loss curves and compute-optimal loss vs. theory on PLRF.** See Figure 25 for details.

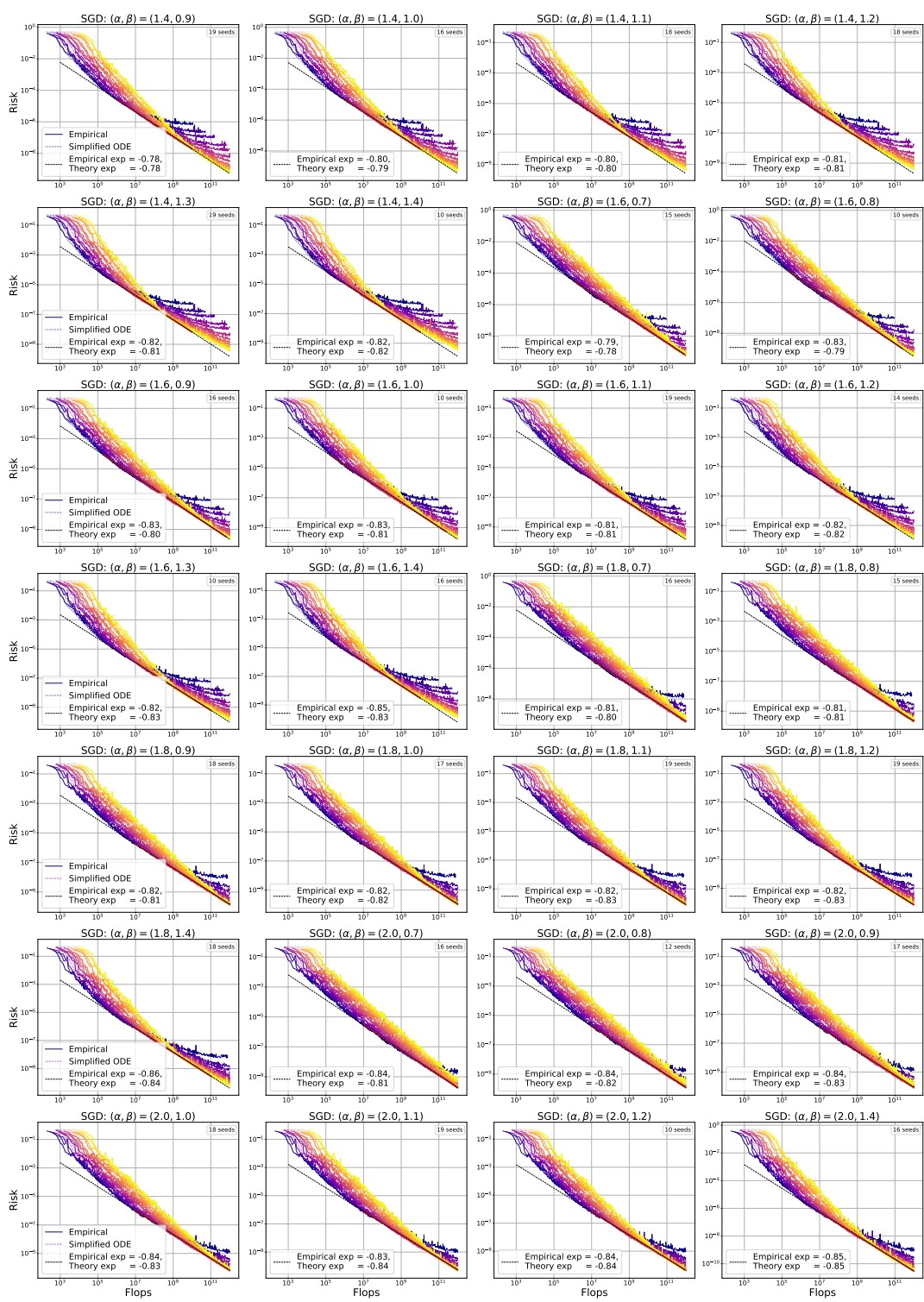

Figure 27: **SGD loss curves and compute-optimal loss vs. theory on PLRF.** See Figure 25 for details.

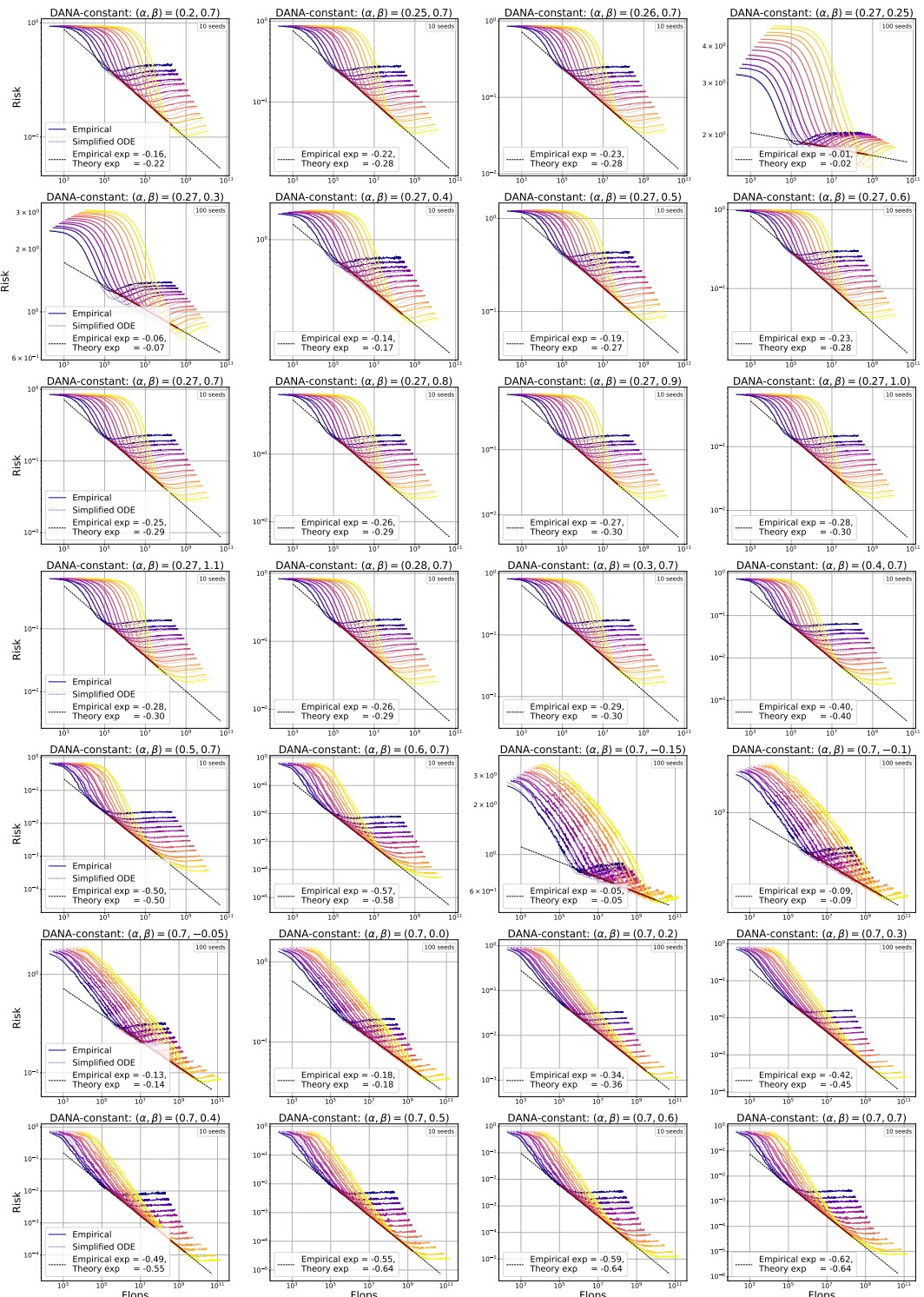

Figure 28: **DANA-constant loss curves and compute-optimal loss vs. theory on PLRF.** For various $(\alpha, \beta)$, we plot the mean population risk for stochastic algorithm runs over 10-100 seeds (see individual figures) for DANA-constant (solid lines) with $\gamma_1 = 1, \gamma_2 = \frac{0.375}{\text{Tr}(D)}, \gamma_3 = \frac{0.1}{d} \times \frac{1}{\text{Tr}(D)}, \Delta(t) = \delta(1+t)^{-1}, \delta = \max\{\frac{2\alpha+2\beta-1}{\alpha}, 2 - \frac{1}{\alpha}\} + 1$. Colors indicate dimensionality $d$ ranging from 200 to 12,800. The stochastic runs of DANA-constant match the solutions of the simplified ODEs in (43) (dotted lines) nearly perfectly. The empirical compute-optimal power-law (dashed black line) is generated using Chinchilla Approach 1 [50] where the solid red highlighted section shows the region fit. The empirical compute-optimal loss exponents nearly match theory for all $(\alpha, \beta)$ tested, with absolute maximum difference of 0.09.

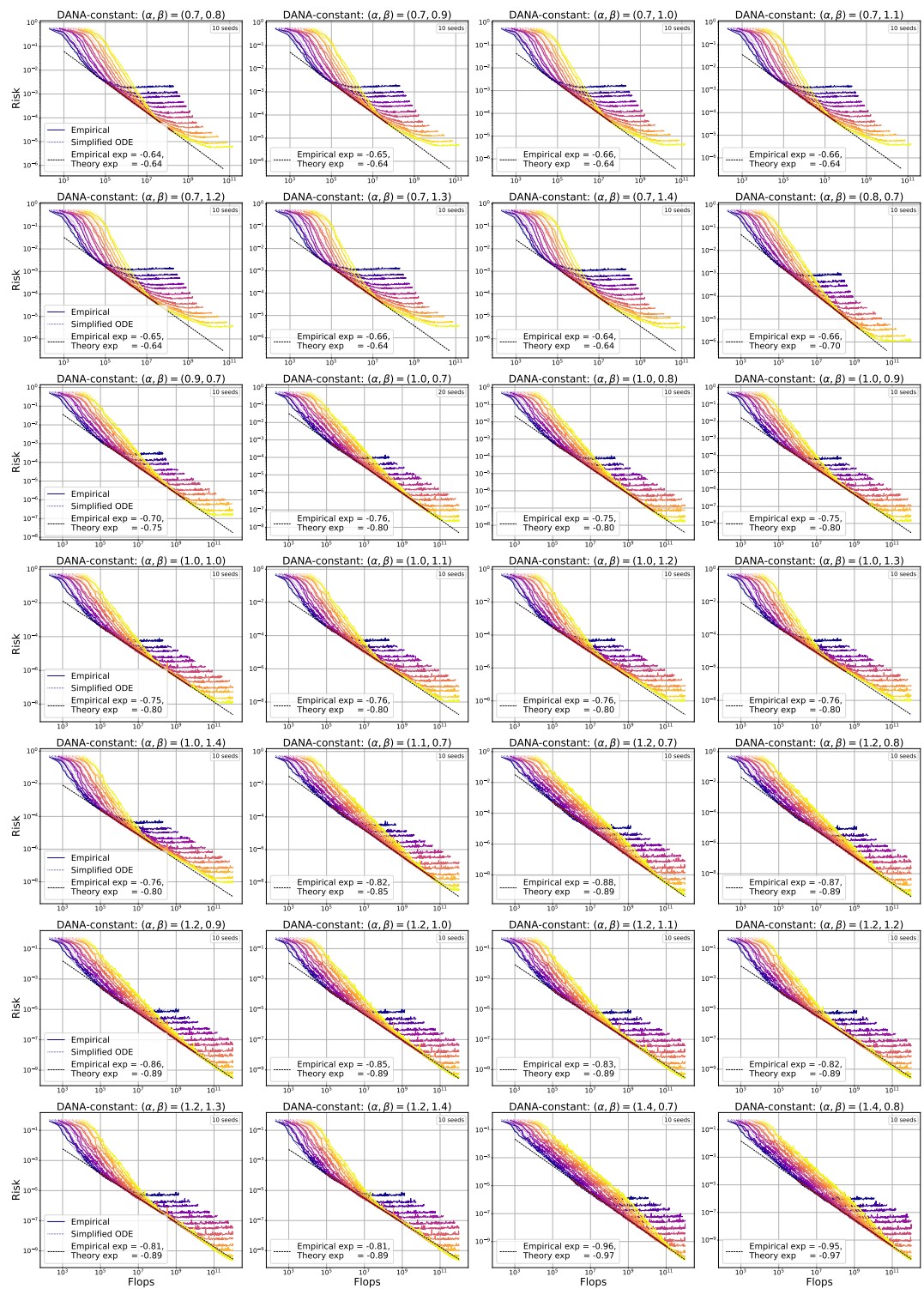

Figure 29: **DANA-constant loss curves and compute-optimal loss vs. theory on PLRF.** See Figure 28 for details.

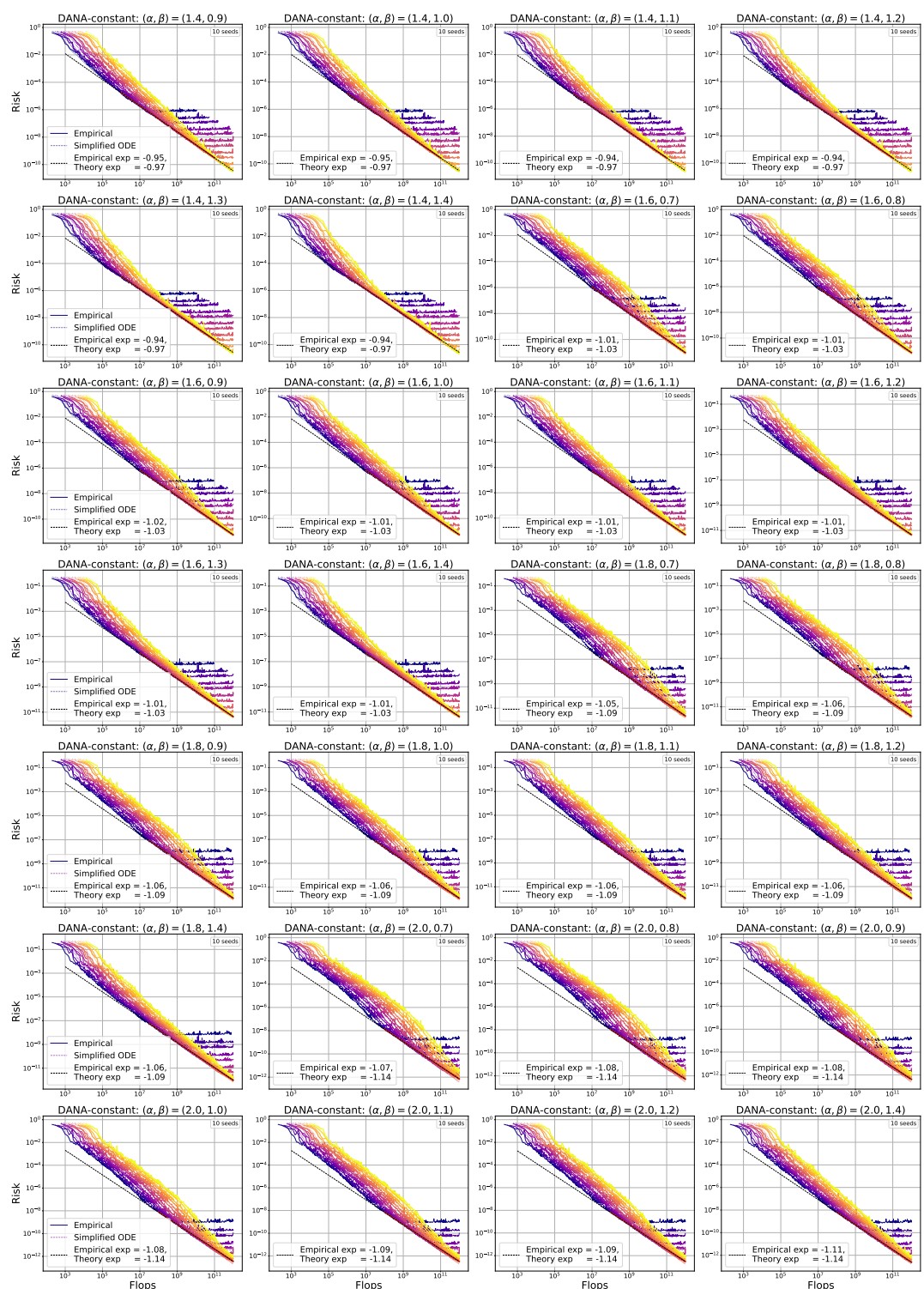

Figure 30: **DANA-constant loss curves and compute-optimal loss vs. theory on PLRF.** See Figure 28 for details.

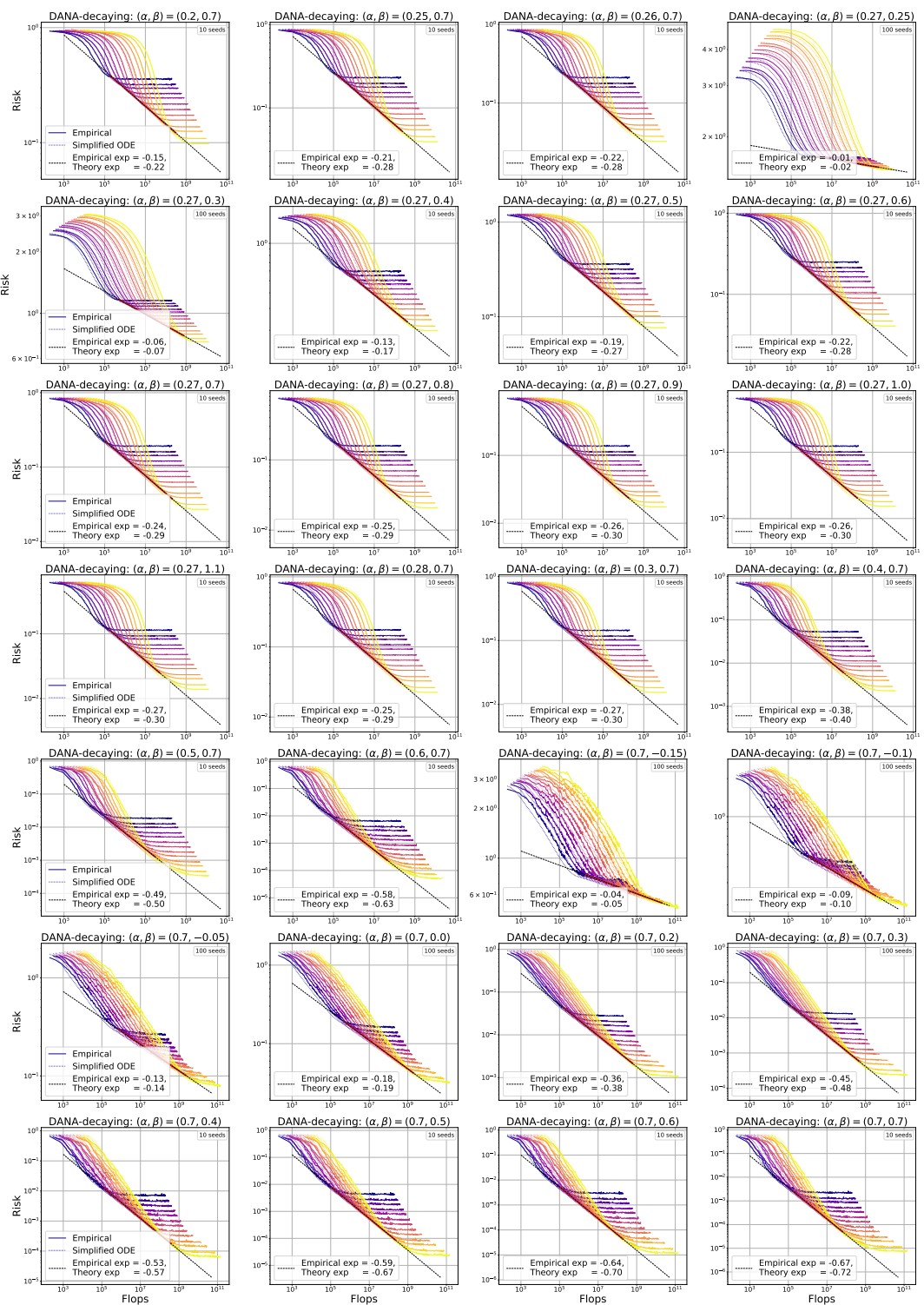

Figure 31: **DANA-decaying loss curves and compute-optimal loss vs. theory on PLRF.** For various $(\alpha, \beta)$, we plot the mean population risk for stochastic algorithm runs over 10-100 seeds (see individual figures) for DANA-decaying (solid lines) with $\gamma_1 = 1, \gamma_2 = \frac{0.375}{\text{Tr}(D)}, \gamma_3(t) = \frac{0.1}{(1+t)^{1/(2\alpha)}}, \Delta(t) = \delta(1+t)^{-1}, \delta = \max\{\frac{2\alpha+2\beta-1}{\alpha}, 2 - \frac{1}{\alpha}\} + 1$. Colors indicate dimensionality $d$ ranging from 200 to 12,800. The stochastic runs of DANA-decaying match the solutions of the simplified ODEs in (43) (dotted lines) nearly perfectly. The empirical compute-optimal power-law (dashed black line) is generated using Chinchilla Approach 1 [50] where the solid red highlighted section shows the region fit. The empirical compute-optimal loss exponents nearly match theory for all $(\alpha, \beta)$ tested, with absolute maximum difference of 0.085.

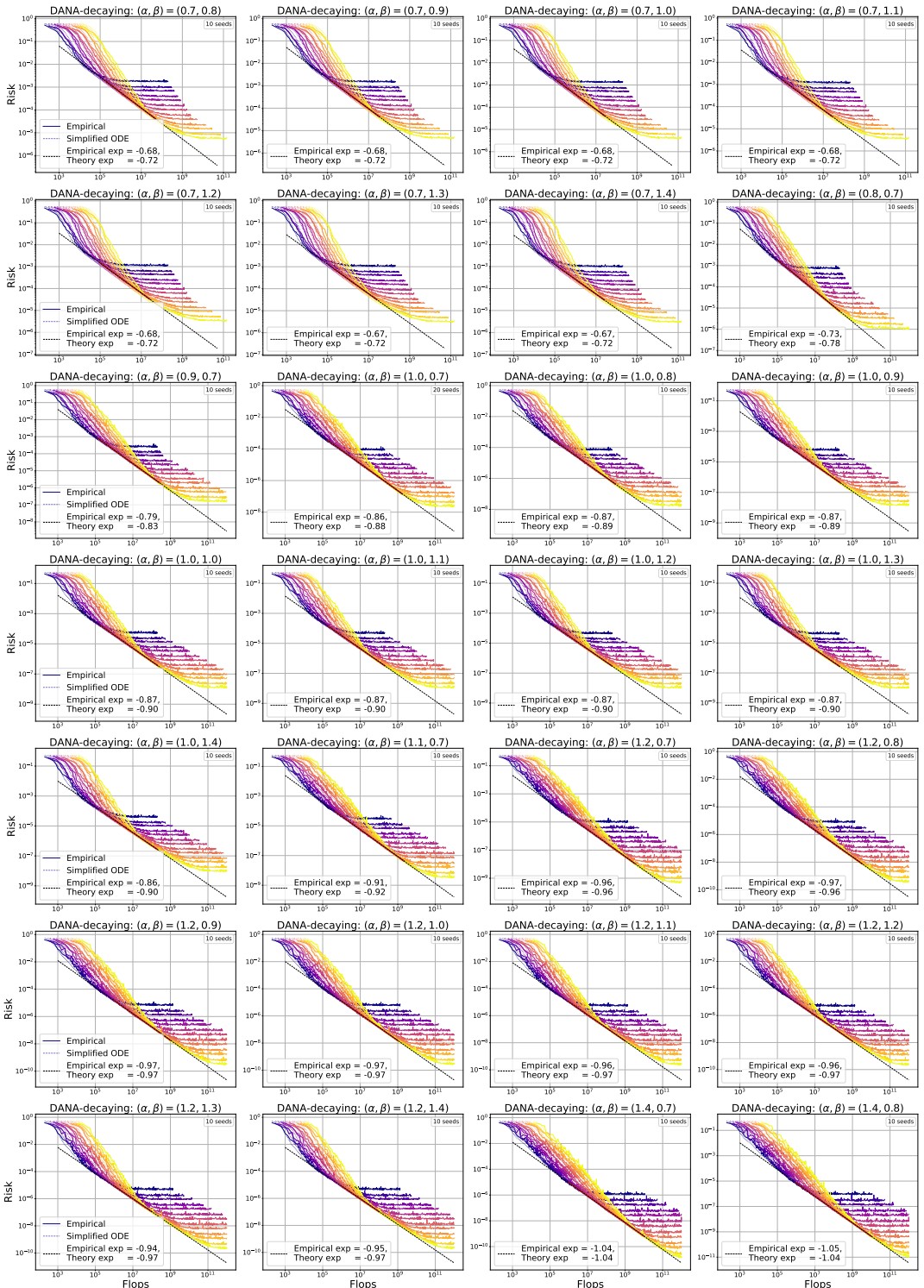

Figure 32: **DANA-decaying loss curves and compute-optimal loss vs. theory on PLRF.** See Figure 31 for details.

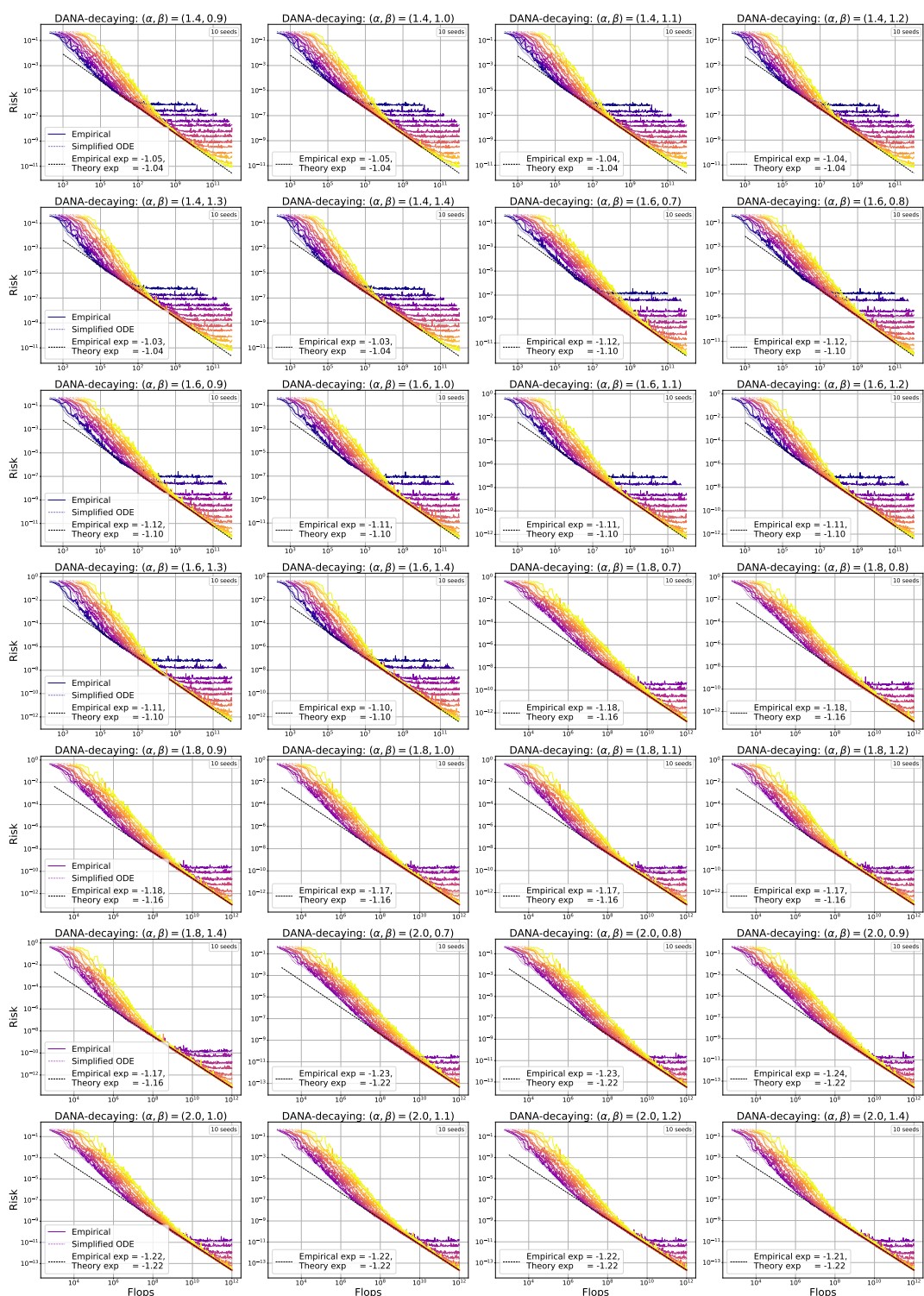

Figure 33: **DANA-decaying loss curves and compute-optimal loss vs. theory on PLRF.** See Figure 31 for details.

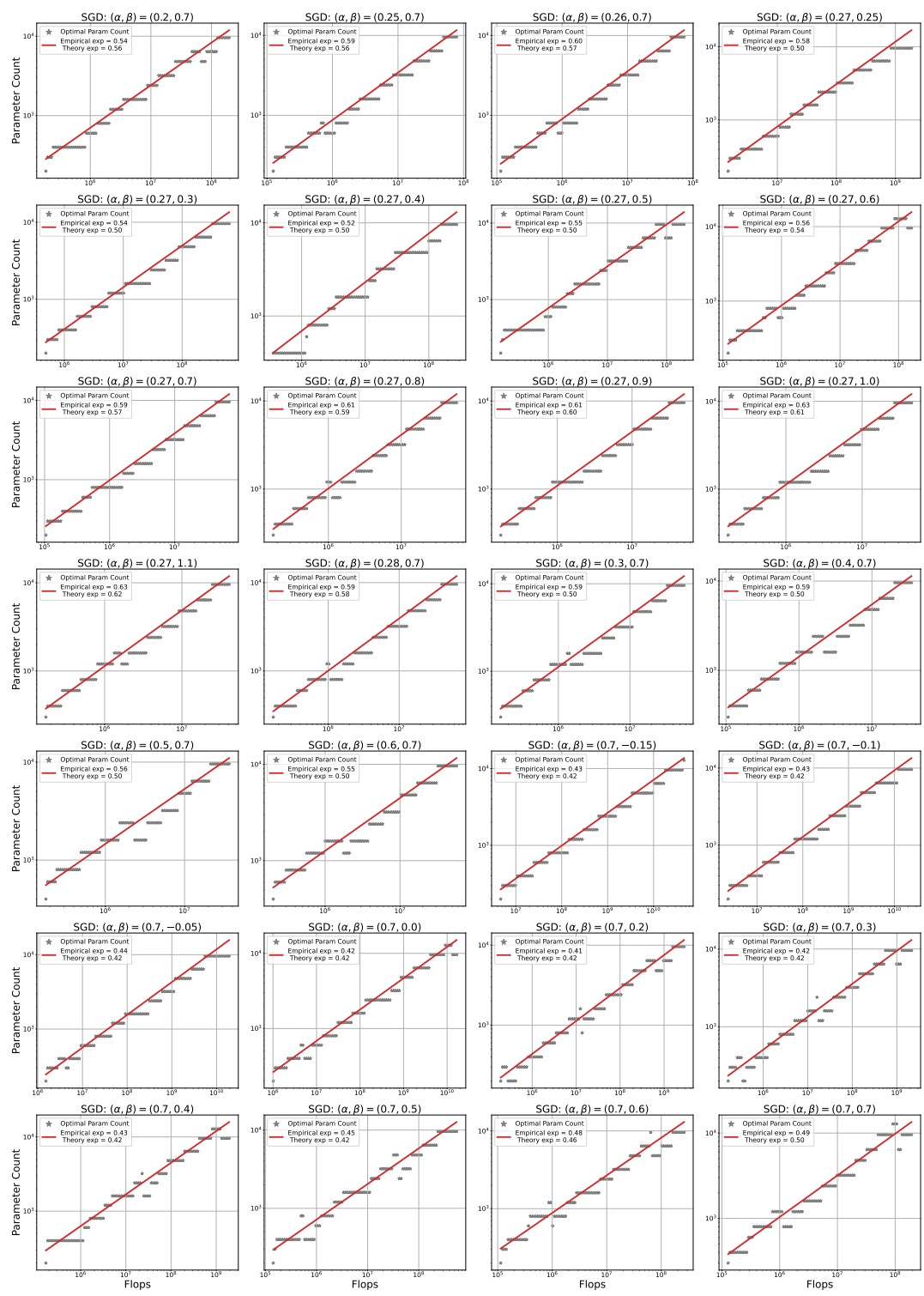

Figure 34: **SGD Chinchilla Approach 1.** Gray stars plot the parameter count of the model size that is optimal for each value of flops using the loss curves from Figure 25. Power laws (red line) fit through these points give the empirical parameter exponent, which matches theoretical predictions within 0.09.

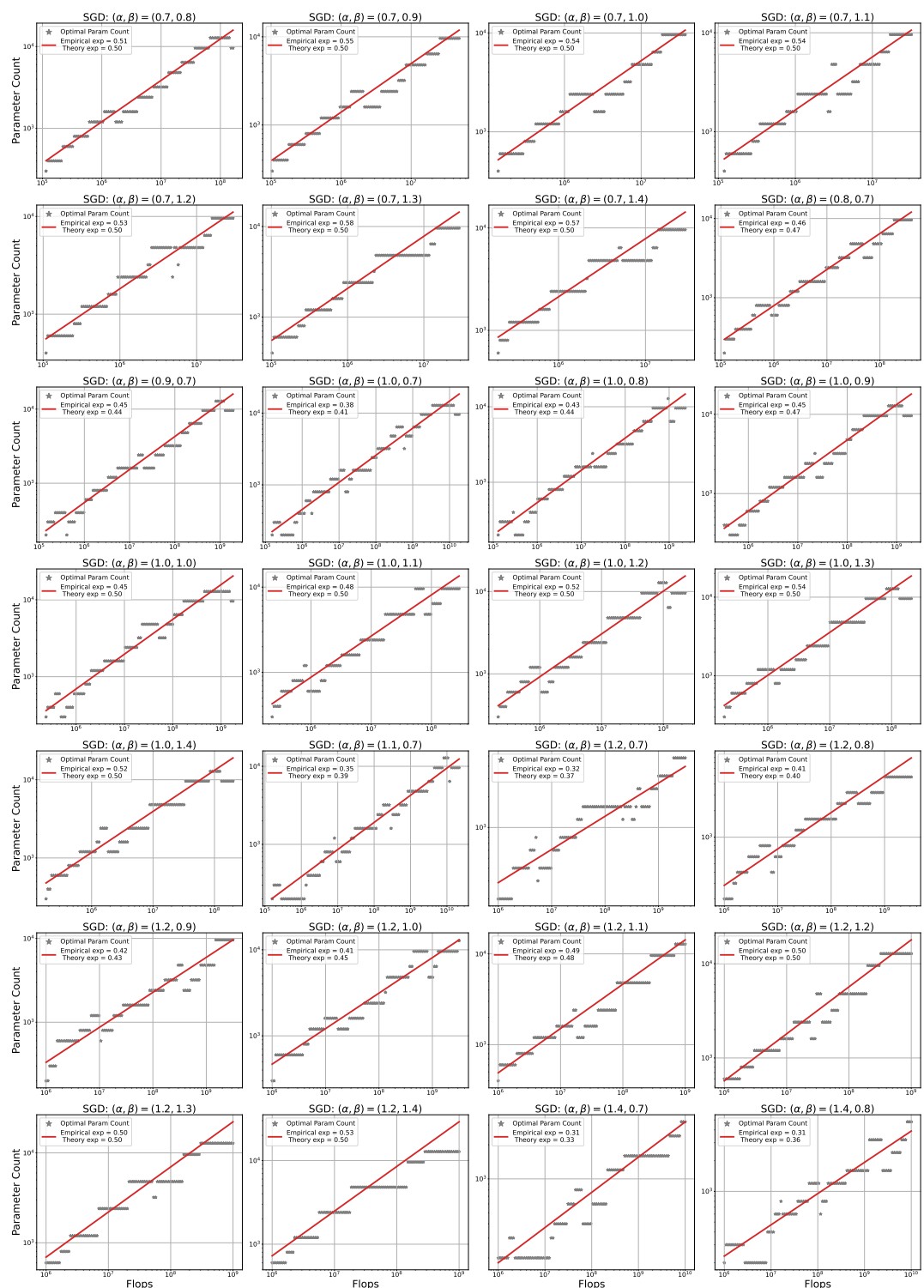

Figure 35: **SGD Chinchilla Approach 1.** Gray stars plot the parameter count of the model size that is optimal for each value of flops using the loss curves from Figure 26. Power laws (red line) fit through these points give the empirical parameter exponent, which matches theoretical predictions within 0.09.

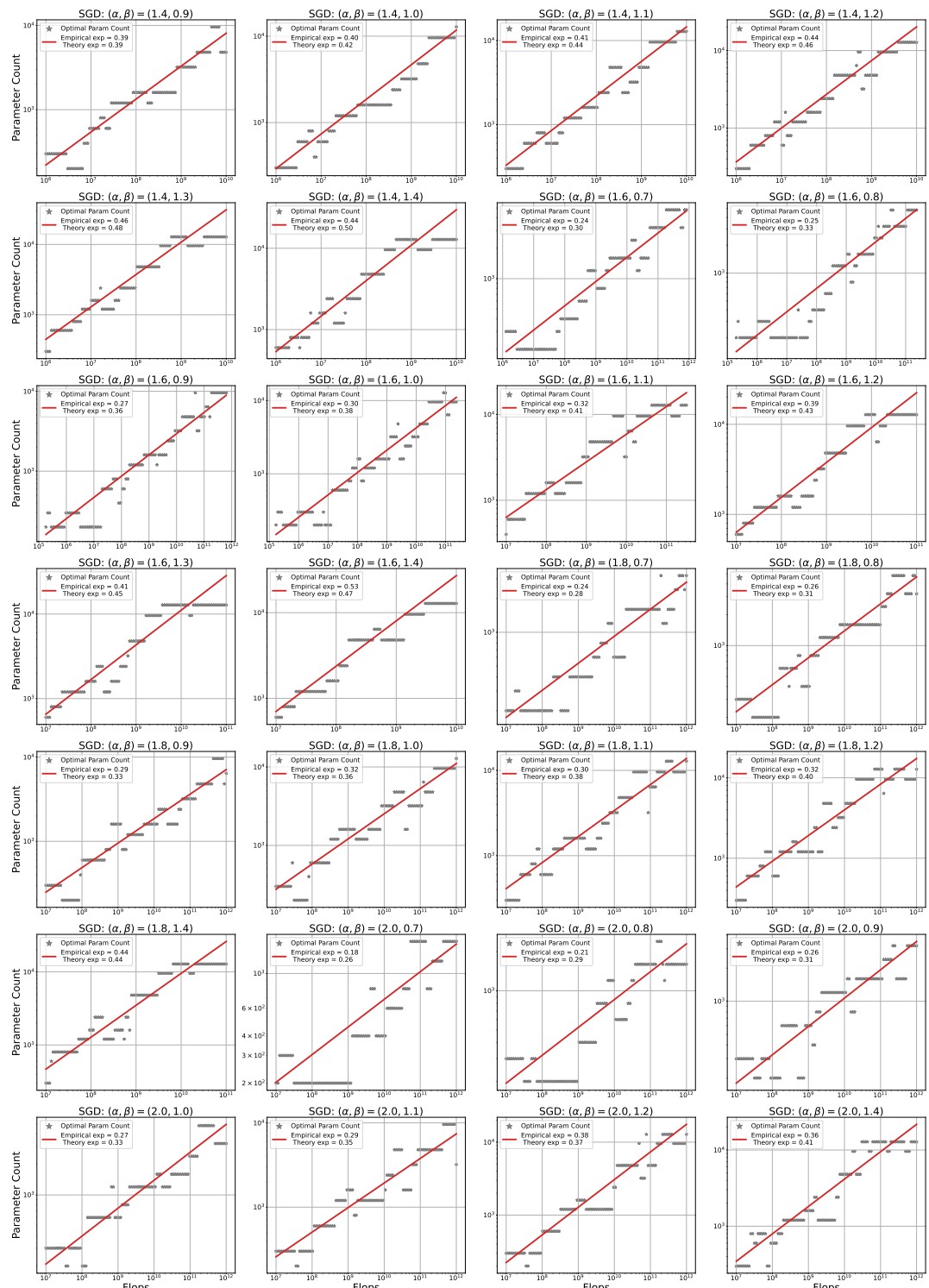

Figure 36: **SGD Chinchilla Approach 1.** Gray stars plot the parameter count of the model size that is optimal for each value of flops using the loss curves from Figure 27. Power laws (red line) fit through these points give the empirical parameter exponent, which matches theoretical predictions within 0.09.

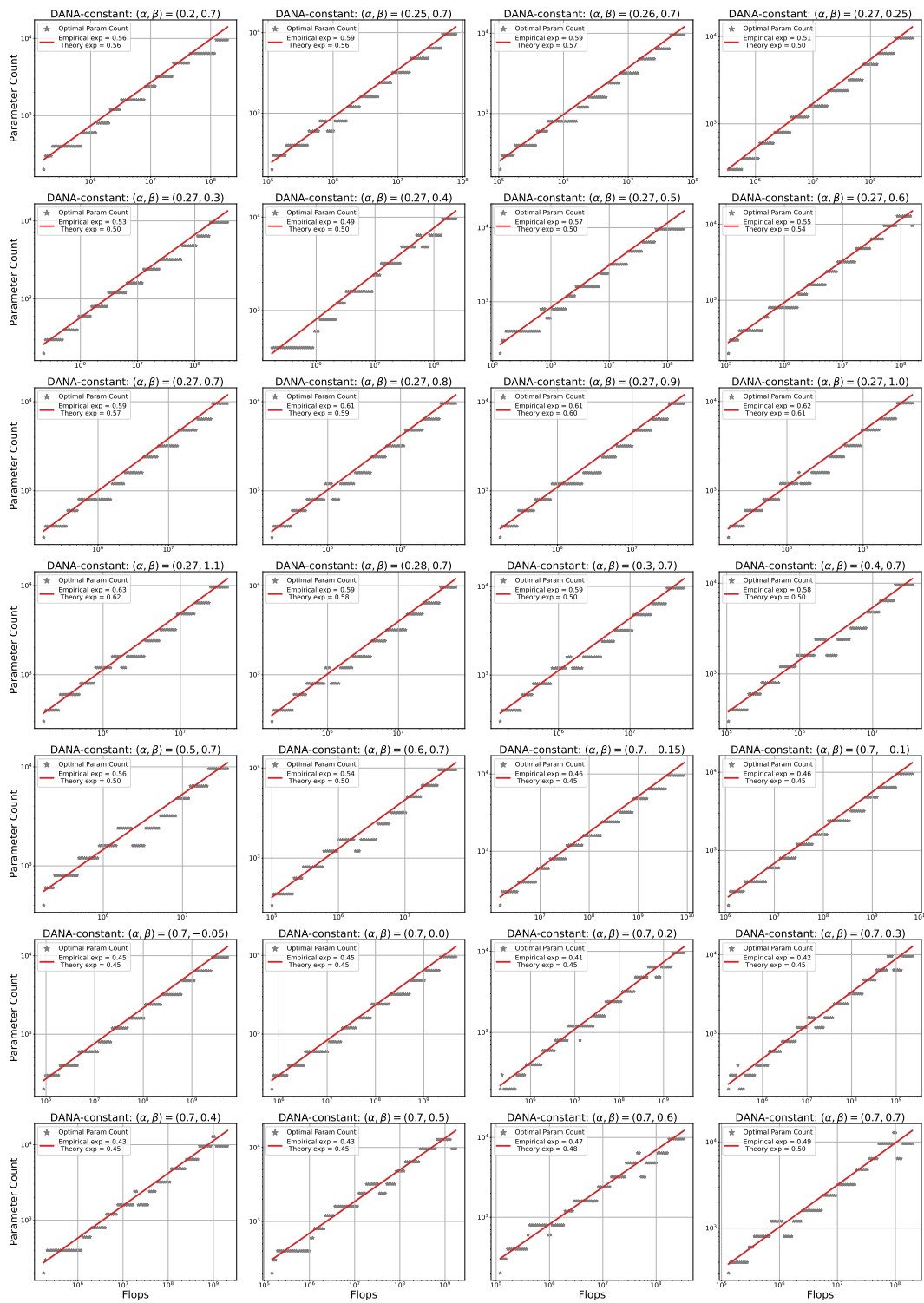

Figure 37: **DANA-constant Chinchilla Approach 1.** Gray stars plot the parameter count of the model size that is optimal for each value of flops using the loss curves from Figure 28. Power laws (red line) fit through these points give the empirical parameter exponent, which matches theoretical predictions within 0.103.

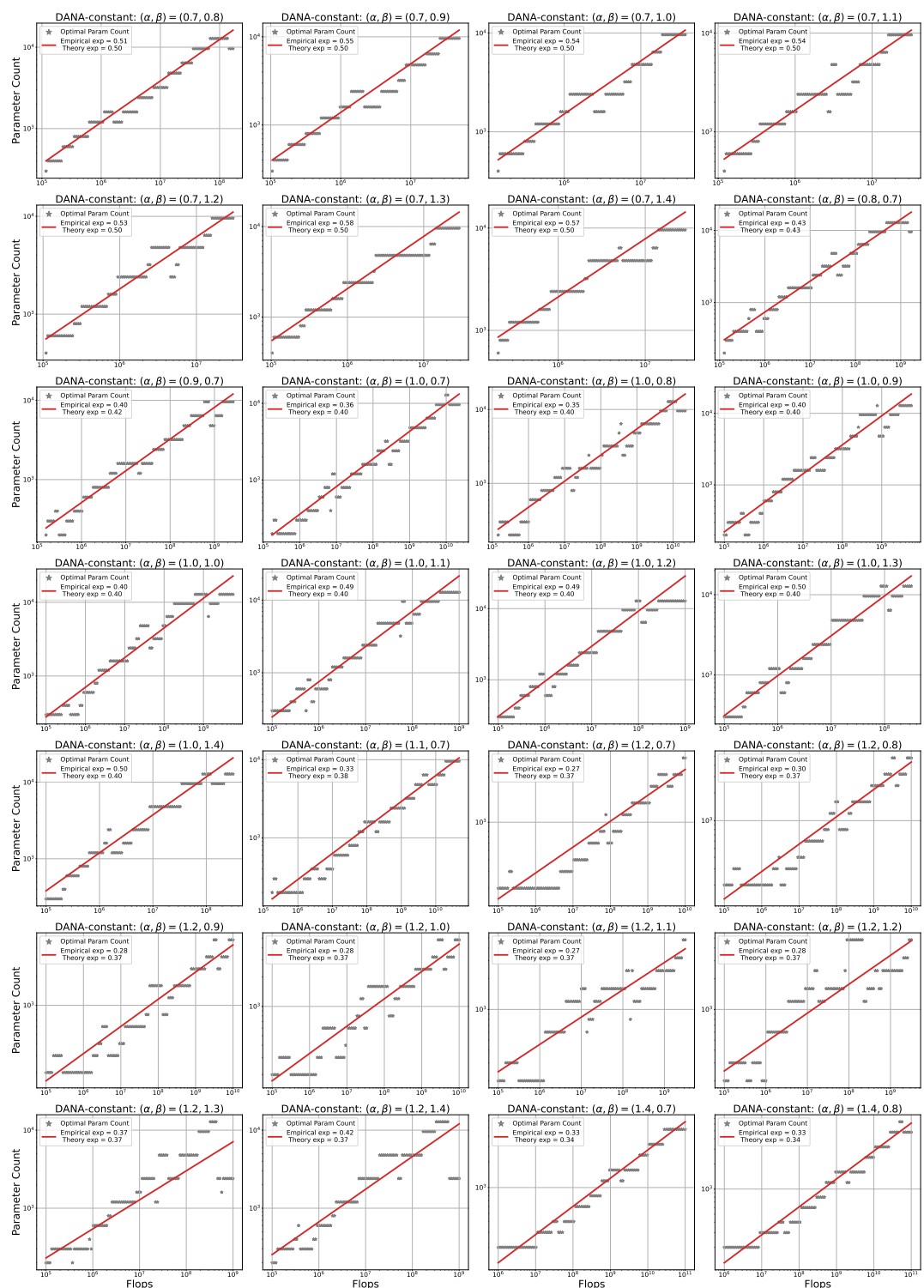

Figure 38: **DANA-constant Chinchilla Approach 1.** Gray stars plot the parameter count of the model size that is optimal for each value of flops using the loss curves from Figure 29. Power laws (red line) fit through these points give the empirical parameter exponent, which matches theoretical predictions within 0.103.

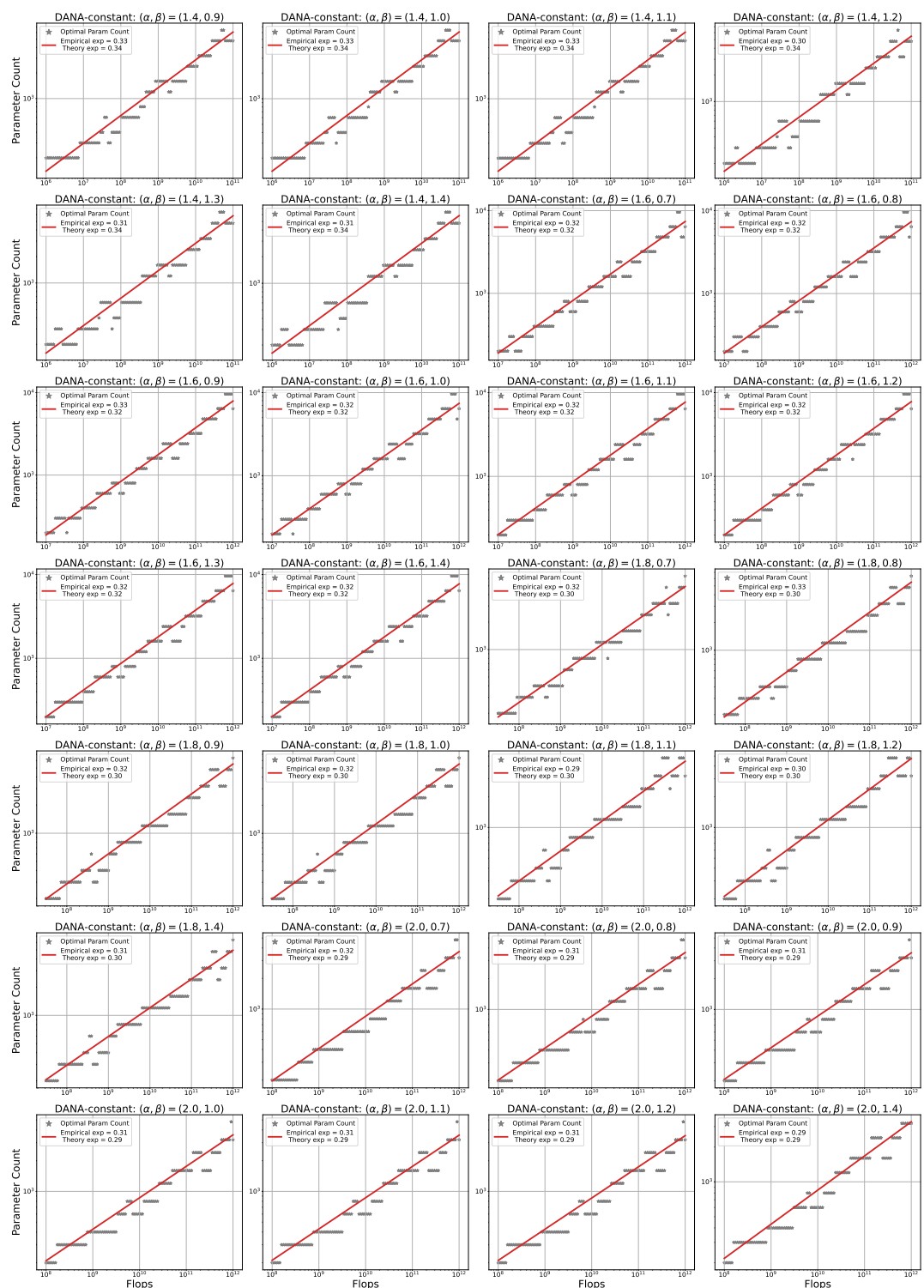

Figure 39: **DANA-constant Chinchilla Approach 1.** Gray stars plot the parameter count of the model size that is optimal for each value of flops using the loss curves from Figure 30. Power laws (red line) fit through these points give the empirical parameter exponent, which matches theoretical predictions within 0.103.

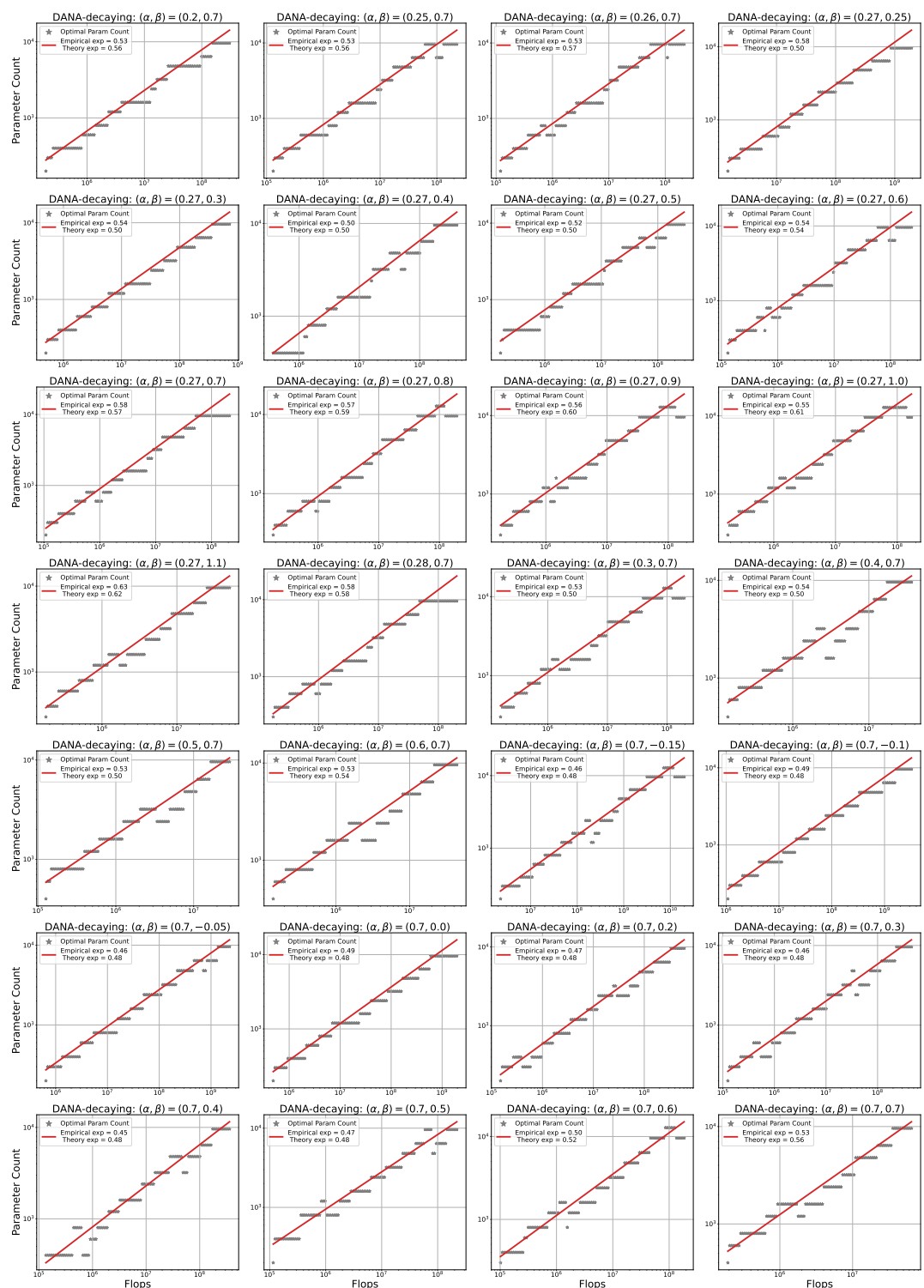

Figure 40: **DANA-decaying Chinchilla Approach 1.** Gray stars plot the parameter count of the model size that is optimal for each value of flops using the loss curves from Figure 31. Power laws (red line) fit through these points give the empirical parameter exponent, which matches theoretical predictions within 0.132.

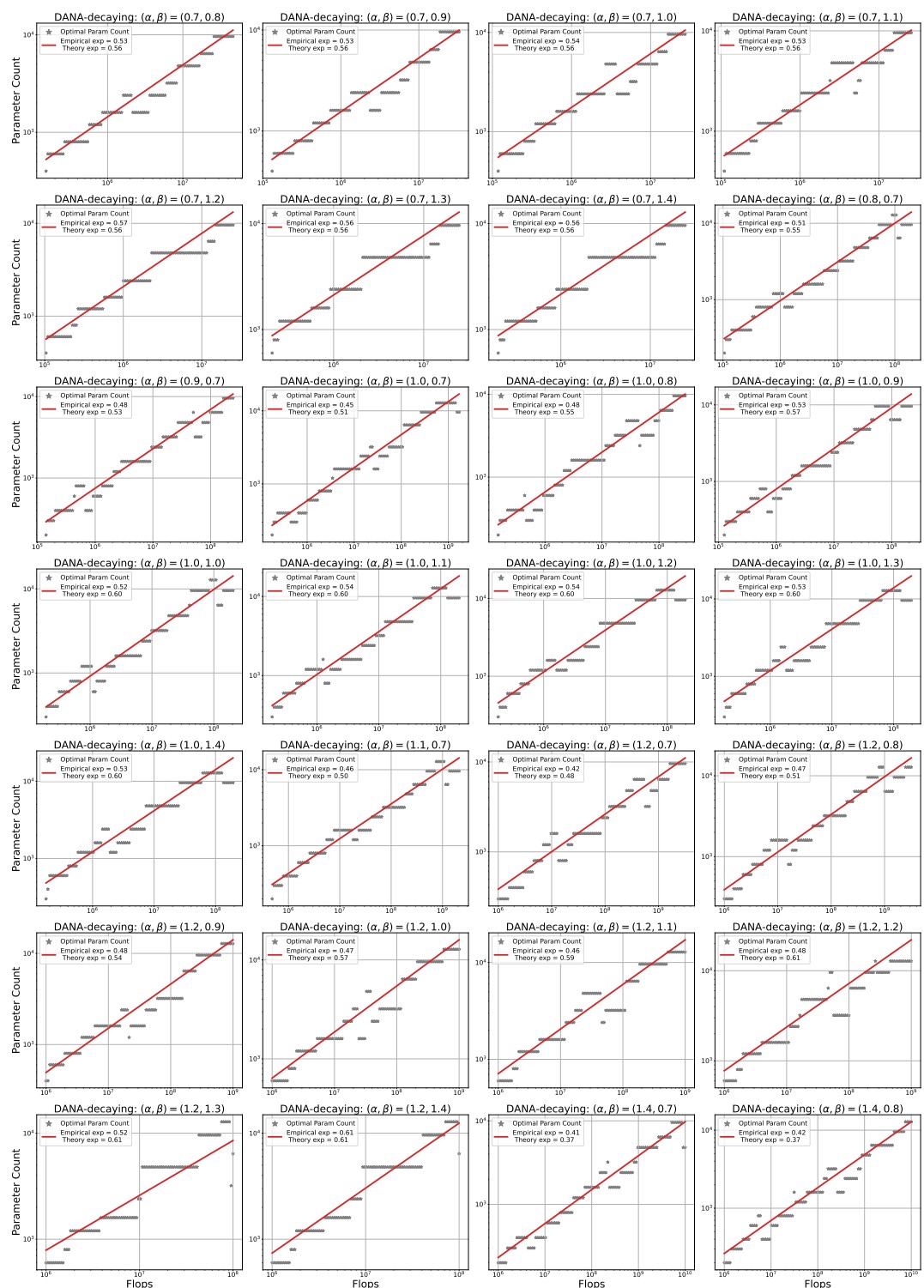

Figure 41: **DANA-decaying Chinchilla Approach 1.** Gray stars plot the parameter count of the model size that is optimal for each value of flops using the loss curves from Figure 32. Power laws (red line) fit through these points give the empirical parameter exponent, which matches theoretical predictions within 0.132.

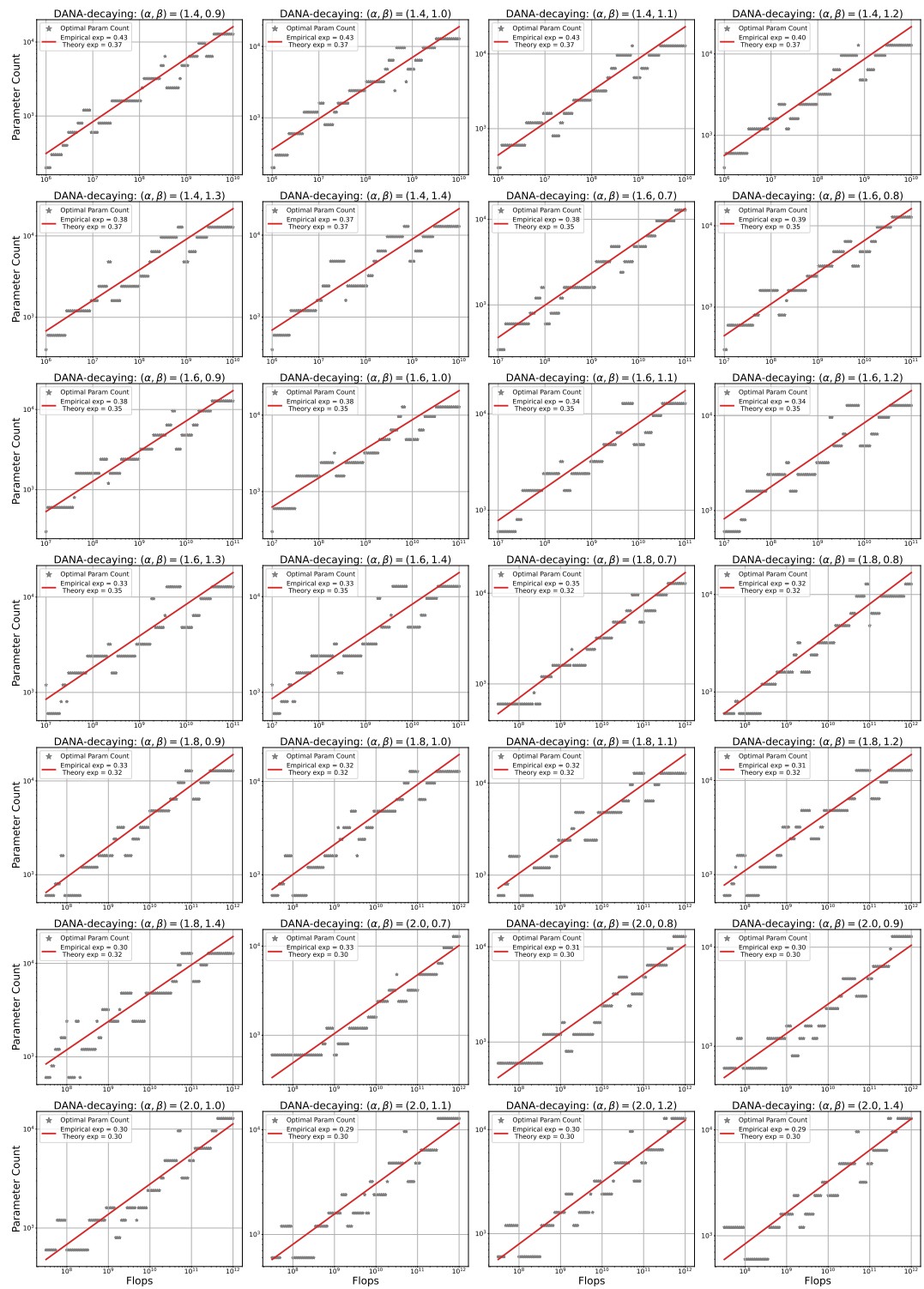

Figure 42: **DANA-decaying Chinchilla Approach 1.** Gray stars plot the parameter count of the model size that is optimal for each value of flops using the loss curves from Figure 33. Power laws (red line) fit through these points give the empirical parameter exponent, which matches theoretical predictions within 0.132.

# NeurIPS Paper Checklist

1. **Claims**

   Question: Do the main claims made in the abstract and introduction accurately reflect the paper's contributions and scope?

   Answer: [Yes]

   Justification: The statements in the abstract and introduction accurately reflect the paper's contributions. The introduction/abstract make clear which of the algorithms we have proven (e.g., we only provide a heuristic for DANA-decaying). We also make it explicit in the statements of the theorems, propositions, etc, the assumptions that we are making. We also verify these assumptions in figures.

   Guidelines:

   - The answer NA means that the abstract and introduction do not include the claims made in the paper.
   - The abstract and/or introduction should clearly state the claims made, including the contributions made in the paper and important assumptions and limitations. A No or NA answer to this question will not be perceived well by the reviewers.
   - The claims made should match theoretical and experimental results, and reflect how much the results can be expected to generalize to other settings.
   - It is fine to include aspirational goals as motivation as long as it is clear that these goals are not attained by the paper.

2. **Limitations**

   Question: Does the paper discuss the limitations of the work performed by the authors?

   Answer: [Yes]

   Justification: At the end of the main paper, we provide a 'Limitations' section. In our statements of propositions, theorems, etc, we make very clear our assumptions. For example, we are explicit that we do create the deterministic system of ODEs directly from the stochastic algorithms but rather SDE (see Section 3). Another example is in Section H, we are clear that we can not solve the system of ODEs in (22) directly instead we solve a simplified system of ODEs and show empirically that the solution to this simplified system of ODEs matches DANA-constant.

   Guidelines:

   - The answer NA means that the paper has no limitation while the answer No means that the paper has limitations, but those are not discussed in the paper.
   - The authors are encouraged to create a separate "Limitations" section in their paper.
   - The paper should point out any strong assumptions and how robust the results are to violations of these assumptions (e.g., independence assumptions, noiseless settings, model well-specification, asymptotic approximations only holding locally). The authors should reflect on how these assumptions might be violated in practice and what the implications would be.
   - The authors should reflect on the scope of the claims made, e.g., if the approach was only tested on a few datasets or with a few runs. In general, empirical results often depend on implicit assumptions, which should be articulated.
   - The authors should reflect on the factors that influence the performance of the approach. For example, a facial recognition algorithm may perform poorly when image resolution is low or images are taken in low lighting. Or a speech-to-text system might not be used reliably to provide closed captions for online lectures because it fails to handle technical jargon.
   - The authors should discuss the computational efficiency of the proposed algorithms and how they scale with dataset size.
   - If applicable, the authors should discuss possible limitations of their approach to address problems of privacy and fairness.

- While the authors might fear that complete honesty about limitations might be used by reviewers as grounds for rejection, a worse outcome might be that reviewers discover limitations that aren't acknowledged in the paper. The authors should use their best judgment and recognize that individual actions in favor of transparency play an important role in developing norms that preserve the integrity of the community. Reviewers will be specifically instructed to not penalize honesty concerning limitations.

3. **Theory assumptions and proofs**

   Question: For each theoretical result, does the paper provide the full set of assumptions and a complete (and correct) proof?

   Answer: [Yes]

   Justification: All theorems, lemmas, etc are proven in the Supplementary Materials. We provide a short outline of the arguments in the main document. In some cases, e.g., DANA-decaying, we only prove for $2\alpha > 1$, but we make this explicitly clear in the main document as well as in Section I. All assumptions are clearly stated in the statements of the theorems. For example, we make it clear that one of our main theorems only holds for $\alpha > 1/4$ and $\alpha + 1 > \beta$. We do provide numerical experiments which show that our results hold beyond this setting.

   Guidelines:

   - The answer NA means that the paper does not include theoretical results.
   - All the theorems, formulas, and proofs in the paper should be numbered and cross-referenced.
   - All assumptions should be clearly stated or referenced in the statement of any theorems.
   - The proofs can either appear in the main paper or the supplemental material, but if they appear in the supplemental material, the authors are encouraged to provide a short proof sketch to provide intuition.
   - Inversely, any informal proof provided in the core of the paper should be complemented by formal proofs provided in appendix or supplemental material.
   - Theorems and Lemmas that the proof relies upon should be properly referenced.

4. **Experimental result reproducibility**

   Question: Does the paper fully disclose all the information needed to reproduce the main experimental results of the paper to the extent that it affects the main claims and/or conclusions of the paper (regardless of whether the code and data are provided or not)?

   Answer: [Yes]

   Justification: In each of the figures, we provide an explicit description of how the image was generated including the numerical set-up. In some cases, due to space limitations, the experimental set-ups are discussed in detail in Section M. We provide a description of the power law random features model (PLRF) in Section 2 and experimental details in Section K which allows for reproducibility of our model on the synthetic data. We also provide explicit description of the set-up for the LSTM experiments in Section L; we include a citation to the datasets that we are using as well.

   Guidelines:

   - The answer NA means that the paper does not include experiments.
   - If the paper includes experiments, a No answer to this question will not be perceived well by the reviewers: Making the paper reproducible is important, regardless of whether the code and data are provided or not.
   - If the contribution is a dataset and/or model, the authors should describe the steps taken to make their results reproducible or verifiable.
   - Depending on the contribution, reproducibility can be accomplished in various ways. For example, if the contribution is a novel architecture, describing the architecture fully might suffice, or if the contribution is a specific model and empirical evaluation, it may be necessary to either make it possible for others to replicate the model with the same dataset, or provide access to the model. In general. releasing code and data is often one good way to accomplish this, but reproducibility can also be provided via detailed instructions for how to replicate the results, access to a hosted model (e.g., in the case

of a large language model), releasing of a model checkpoint, or other means that are appropriate to the research performed.

- While NeurIPS does not require releasing code, the conference does require all submissions to provide some reasonable avenue for reproducibility, which may depend on the nature of the contribution. For example
  - (a) If the contribution is primarily a new algorithm, the paper should make it clear how to reproduce that algorithm.
  - (b) If the contribution is primarily a new model architecture, the paper should describe the architecture clearly and fully.
  - (c) If the contribution is a new model (e.g., a large language model), then there should either be a way to access this model for reproducing the results or a way to reproduce the model (e.g., with an open-source dataset or instructions for how to construct the dataset).
  - (d) We recognize that reproducibility may be tricky in some cases, in which case authors are welcome to describe the particular way they provide for reproducibility. In the case of closed-source models, it may be that access to the model is limited in some way (e.g., to registered users), but it should be possible for other researchers to have some path to reproducing or verifying the results.

5. **Open access to data and code**

Question: Does the paper provide open access to the data and code, with sufficient instructions to faithfully reproduce the main experimental results, as described in supplemental material?

Answer: [NA]

Justification: The paper does not include experiments that require significant code. The model we analyze is a simple power law random features model (PLRF) applied to synthetic data. As such, the code can be readily produced by following the set-up seen in the captions and/or our numerical simulations section, Section K. The model has been used before in other papers. Additionally for the LSTM experiments, we are using a set-up similar to other papers [54] and reproducing Fig. 2 with DANA-decaying and the C4 text dataset [84].

Guidelines:

- The answer NA means that paper does not include experiments requiring code.
- Please see the NeurIPS code and data submission guidelines (https://nips.cc/public/guides/CodeSubmissionPolicy) for more details.
- While we encourage the release of code and data, we understand that this might not be possible, so 'No' is an acceptable answer. Papers cannot be rejected simply for not including code, unless this is central to the contribution (e.g., for a new open-source benchmark).
- The instructions should contain the exact command and environment needed to run to reproduce the results. See the NeurIPS code and data submission guidelines (https://nips.cc/public/guides/CodeSubmissionPolicy) for more details.
- The authors should provide instructions on data access and preparation, including how to access the raw data, preprocessed data, intermediate data, and generated data, etc.
- The authors should provide scripts to reproduce all experimental results for the new proposed method and baselines. If only a subset of experiments are reproducible, they should state which ones are omitted from the script and why.
- At submission time, to preserve anonymity, the authors should release anonymized versions (if applicable).
- Providing as much information as possible in supplemental material (appended to the paper) is recommended, but including URLs to data and code is permitted.

6. **Experimental setting/details**

Question: Does the paper specify all the training and test details (e.g., data splits, hyper-parameters, how they were chosen, type of optimizer, etc.) necessary to understand the results?

Answer: [Yes]

Justification: The experimental design, including the power law exponents, hyperparameters, fixed stepsizes, choices of $d$ and $v$, and the numerical simulations for solving the ODEs are all written in the captions of the figures and/or Section K, Section L or Section M. We also intend to release the code for numerically computing the ODEs.

Guidelines:

- The answer NA means that the paper does not include experiments.
- The experimental setting should be presented in the core of the paper to a level of detail that is necessary to appreciate the results and make sense of them.
- The full details can be provided either with the code, in appendix, or as supplemental material.

7. **Experiment statistical significance**

Question: Does the paper report error bars suitably and correctly defined or other appropriate information about the statistical significance of the experiments?

Answer: [Yes]

Justification: We record how we generate the empirical compute-optimal exponents using statistical tools that were first deployed in other papers such as [50]. We report the number of random seeds used for each PLRF experiment in Section K. We report $R^2$ values for all LSTM loss exponents. We are careful to explain when and why the theory deviates from the numerical simulations. Often this is due to finite $d$ and $v$ effects and the slow behavior of the theory to the asymptotics.

Guidelines:

- The answer NA means that the paper does not include experiments.
- The authors should answer "Yes" if the results are accompanied by error bars, confidence intervals, or statistical significance tests, at least for the experiments that support the main claims of the paper.
- The factors of variability that the error bars are capturing should be clearly stated (for example, train/test split, initialization, random drawing of some parameter, or overall run with given experimental conditions).
- The method for calculating the error bars should be explained (closed form formula, call to a library function, bootstrap, etc.)
- The assumptions made should be given (e.g., Normally distributed errors).
- It should be clear whether the error bar is the standard deviation or the standard error of the mean.
- It is OK to report 1-sigma error bars, but one should state it. The authors should preferably report a 2-sigma error bar than state that they have a 96% CI, if the hypothesis of Normality of errors is not verified.
- For asymmetric distributions, the authors should be careful not to show in tables or figures symmetric error bars that would yield results that are out of range (e.g. negative error rates).
- If error bars are reported in tables or plots, The authors should explain in the text how they were calculated and reference the corresponding figures or tables in the text.

8. **Experiments compute resources**

Question: For each experiment, does the paper provide sufficient information on the computer resources (type of compute workers, memory, time of execution) needed to reproduce the experiments?

Answer: [Yes]

Justification: As this paper is about compute-optimal curves, we provide details on the exact number of flops required to perform the experiments in Section K and Section L. The compute resources are also well known in the community using the standard $6ND$ heuristic [50, 54] for flops.

Guidelines:

- The answer NA means that the paper does not include experiments.

- The paper should indicate the type of compute workers CPU or GPU, internal cluster, or cloud provider, including relevant memory and storage.
- The paper should provide the amount of compute required for each of the individual experimental runs as well as estimate the total compute.
- The paper should disclose whether the full research project required more compute than the experiments reported in the paper (e.g., preliminary or failed experiments that didn't make it into the paper).

9. **Code of ethics**

Question: Does the research conducted in the paper conform, in every respect, with the NeurIPS Code of Ethics https://neurips.cc/public/EthicsGuidelines?

Answer: [Yes]

Justification: We have reviewed the code of ethics. We have made our utmost attempt to adhere to the guidelines provided by NeurIPS. We do not use any human subjects nor any datasets. We did our best to cite all the relevent related work. Given that our work is in the foundational research, it is difficult to mitigate all the risks as the downstream effects of theory are long, but we have done our best. The model is completely synthetic using the standard stochastic momentum-type algorithms; thus we don't, to the best of our knowledge, anticipate any risks. We have included a "Broader Impact" statement at the beginning of the "Supplemental Materials."

Guidelines:

- The answer NA means that the authors have not reviewed the NeurIPS Code of Ethics.
- If the authors answer No, they should explain the special circumstances that require a deviation from the Code of Ethics.
- The authors should make sure to preserve anonymity (e.g., if there is a special consideration due to laws or regulations in their jurisdiction).

10. **Broader impacts**

Question: Does the paper discuss both potential positive societal impacts and negative societal impacts of the work performed?

Answer: [NA]

Justification: The work presented is purely foundational research and is not directly tied to any particular application. We study a simple random features model with power law data and target and we solve the model using a common algorithm SGD. Given the theoretical nature of this work, we do not anticipate any direct ethical and societal issues. See our Broader Impact Statement in the appendix.

Guidelines:

- The answer NA means that there is no societal impact of the work performed.
- If the authors answer NA or No, they should explain why their work has no societal impact or why the paper does not address societal impact.
- Examples of negative societal impacts include potential malicious or unintended uses (e.g., disinformation, generating fake profiles, surveillance), fairness considerations (e.g., deployment of technologies that could make decisions that unfairly impact specific groups), privacy considerations, and security considerations.
- The conference expects that many papers will be foundational research and not tied to particular applications, let alone deployments. However, if there is a direct path to any negative applications, the authors should point it out. For example, it is legitimate to point out that an improvement in the quality of generative models could be used to generate deepfakes for disinformation. On the other hand, it is not needed to point out that a generic algorithm for optimizing neural networks could enable people to train models that generate Deepfakes faster.
- The authors should consider possible harms that could arise when the technology is being used as intended and functioning correctly, harms that could arise when the technology is being used as intended but gives incorrect results, and harms following from (intentional or unintentional) misuse of the technology.

- If there are negative societal impacts, the authors could also discuss possible mitigation strategies (e.g., gated release of models, providing defenses in addition to attacks, mechanisms for monitoring misuse, mechanisms to monitor how a system learns from feedback over time, improving the efficiency and accessibility of ML).

11. **Safeguards**

Question: Does the paper describe safeguards that have been put in place for responsible release of data or models that have a high risk for misuse (e.g., pretrained language models, image generators, or scraped datasets)?

Answer: [NA]

Justification: The paper poses no such risks. The majority of the work is theoretical and focuses on the power-law random features using only synthetic data generated from a normal distribution. This model is a standard statistical model (e.g., least squares) which is a textbook learning problem. The LSTM experiments serve only to illustrate the theoretical results act as a proxy in a more realistic setting, and are trained on a public language dataset (C4,[84]). We do not release any data or pretrained models.

Guidelines:

- The answer NA means that the paper poses no such risks.
- Released models that have a high risk for misuse or dual-use should be released with necessary safeguards to allow for controlled use of the model, for example by requiring that users adhere to usage guidelines or restrictions to access the model or implementing safety filters.
- Datasets that have been scraped from the Internet could pose safety risks. The authors should describe how they avoided releasing unsafe images.
- We recognize that providing effective safeguards is challenging, and many papers do not require this, but we encourage authors to take this into account and make a best faith effort.

12. **Licenses for existing assets**

Question: Are the creators or original owners of assets (e.g., code, data, models), used in the paper, properly credited and are the license and terms of use explicitly mentioned and properly respected?

Answer: [Yes]

Justification: We acknowledge via citations that the model we study was introduced before by others (e.g., Maloney, Roberts, and Sully paper). We cite the datasets we are using, e.g., C4 language dataset [84] and software use [18]. We include an extensive relate work section in Section A where we provide tables comparing sample complexity across multiple algorithms.

Guidelines:

- The answer NA means that the paper does not use existing assets.
- The authors should cite the original paper that produced the code package or dataset.
- The authors should state which version of the asset is used and, if possible, include a URL.
- The name of the license (e.g., CC-BY 4.0) should be included for each asset.
- For scraped data from a particular source (e.g., website), the copyright and terms of service of that source should be provided.
- If assets are released, the license, copyright information, and terms of use in the package should be provided. For popular datasets, `paperswithcode.com/datasets` has curated licenses for some datasets. Their licensing guide can help determine the license of a dataset.
- For existing datasets that are re-packaged, both the original license and the license of the derived asset (if it has changed) should be provided.
- If this information is not available online, the authors are encouraged to reach out to the asset's creators.

13. **New assets**

Question: Are new assets introduced in the paper well documented and is the documentation provided alongside the assets?

Answer: [NA]

Justification: We do not intend to release any new assets. The work is a theoretical analysis of random features model along with limited language model experiments that illustrate the theoretical behavior. We will release some code for solving the ODEs.

Guidelines:

- The answer NA means that the paper does not release new assets.
- Researchers should communicate the details of the dataset/code/model as part of their submissions via structured templates. This includes details about training, license, limitations, etc.
- The paper should discuss whether and how consent was obtained from people whose asset is used.
- At submission time, remember to anonymize your assets (if applicable). You can either create an anonymized URL or include an anonymized zip file.

14. **Crowdsourcing and research with human subjects**

Question: For crowdsourcing experiments and research with human subjects, does the paper include the full text of instructions given to participants and screenshots, if applicable, as well as details about compensation (if any)?

Answer: [NA]

Justification: The paper does not involve crowdsourcing nor research with human subjects. The work is purely theoretical on a simple model.

Guidelines:

- The answer NA means that the paper does not involve crowdsourcing nor research with human subjects.
- Including this information in the supplemental material is fine, but if the main contribution of the paper involves human subjects, then as much detail as possible should be included in the main paper.
- According to the NeurIPS Code of Ethics, workers involved in data collection, curation, or other labor should be paid at least the minimum wage in the country of the data collector.

15. **Institutional review board (IRB) approvals or equivalent for research with human subjects**

Question: Does the paper describe potential risks incurred by study participants, whether such risks were disclosed to the subjects, and whether Institutional Review Board (IRB) approvals (or an equivalent approval/review based on the requirements of your country or institution) were obtained?

Answer: [NA]

Justification: The work does not involve crowdsourcing nor research with human subjects. The data used in this work is generated synthetically.

Guidelines:

- The answer NA means that the paper does not involve crowdsourcing nor research with human subjects.
- Depending on the country in which research is conducted, IRB approval (or equivalent) may be required for any human subjects research. If you obtained IRB approval, you should clearly state this in the paper.
- We recognize that the procedures for this may vary significantly between institutions and locations, and we expect authors to adhere to the NeurIPS Code of Ethics and the guidelines for their institution.
- For initial submissions, do not include any information that would break anonymity (if applicable), such as the institution conducting the review.

16. **Declaration of LLM usage**

Question: Does the paper describe the usage of LLMs if it is an important, original, or non-standard component of the core methods in this research? Note that if the LLM is used only for writing, editing, or formatting purposes and does not impact the core methodology, scientific rigorousness, or originality of the research, declaration is not required.

Answer: [NA]

Justification: The core development of the research in this paper does not involve LLM as any important, original, or non-standard components. LLMs were used to clean up some of the grammar aspects as well as make our code more efficient, but the ideas and proofs (core of this work) did not involve any LLMs.

Guidelines:

- The answer NA means that the core method development in this research does not involve LLMs as any important, original, or non-standard components.
- Please refer to our LLM policy (`https://neurips.cc/Conferences/2025/LLM`) for what should or should not be described.

