# OpenReview forum: "Dimension-adapted Momentum Outscales SGD"
_NeurIPS.cc/2025/Conference — NeurIPS 2025 spotlight_

### Official Review · Reviewer_NJzg · 2025-07-02

**Clarity:** 3
**Significance:** 4
**Originality:** 3
**Rating:** 5
**Confidence:** 3

**Summary:**

The authors study the scaling laws for stochastic momentum algorithms for random feature models with the input
distribution and ground truth direction satisfying power laws. They show via both theoretical analysis (under certain
simplifying assumptions) and experiments that (a) SGD with momentum yields the same scaling law exponent as SGD, and
(b) the scaling law exponent can be improved if we allow the momentum learning rate to depend on the dimension
(or the time-dependent effective dimension).

**Questions:**

* In Assumption 1, you implicitly assume that the input distribution and the ground truth direction share the same
  eigenvectors. Is this essential? What will happen if they are not completely aligned?
* Is your rates optimal in any sense? (Is there a way to define optimal in this setting, that is more fine-grained than the minimax rate?)

**Ethical Concerns:**

["NO or VERY MINOR ethics concerns only"]

**Final Justification:**

I believe this is an important paper as it proves that, at least in certain cases, the scaling law exponent can be changed by changing the algorithm and contains extensive experiments to support their theoretical results.
The authors provided a proof sketch in the rebuttal. It looks reasonable to me and this resolves my main concern.

**Limitations:**

The authors suggest in the introduction that the theoretical analysis is rigorous (line 30-32), while they actually
need to ignore some higher-order terms and use the deterministic equivalent (which is only mentioned in one sentence
around line 131) at certain point of the analysis. To be fair, the authors do provide experimental results to justify
these approximations, but they should be more clear about this non-rigorousness.

**Quality:**

3

**Strengths And Weaknesses:**

### Strengths

This seems to be an important paper, as it shows that at least in certain cases, changing the algorithm, in particular,
letting the momentum parameter depend on the (effective) dimension, can potentially change the exponent of the scaling law,
instead of just the constant. Moreover, the authors conduct extensive theoretical analysis and experiments, covering a
wide range of possible power law random feature parameters, algorithms, and training regimes, to support their claims.

The framework (Gen-Mom-SGD) the authors use covers many existing algorithms as special cases and might be of
independent interest. In particular, this framework/strategy allows continuous-analysis of momentum-based algorithms,
which is often quite tricky.

### Weakness

One issue of this paper is the lack of a proof sketch section. The main text is mostly about the results, experiments,
and interpretation. Without a proof sketch section (which can be put in the appendix), it is extremely hard to check the
soundness of the proof of a 194 pages long paper.

---

> ### Author Rebuttal · Authors · 2025-07-30
>
> We thank the reviewer for their time, constructive feedback, and positive evaluation of our work. We are heartened to hear that the reviewer feels that our results on changing the exponent make “an important paper”, and that our theoretical and empirical evaluation was “extensive”.  We address below the key clarification points raised by the reviewer.
>
> **Lack of a proof sketch section**
>
> We acknowledge the reviewer’s legitimate concern about the lack of a proof sketch section. We provide below a detailed proof sketch which we will add to the updated manuscript.
>
> *Detailed proof sketch:*
>
> I. The first step is to derive an evolution equation for the quadratic risk using the raw algorithm updates Gen-Mom-SGD.
> 1. We adopt in section C.1 a continuized analysis in (Gen-Mom-SGD) by replacing the update times by the jumps of a Poisson process of rate $1$ (eq. 17-18). This allows us to reduce the analysis to ODE systems without sending the learning rates to $0$.
> 2. In Sec. C.1.1 we derive ODEs for the projected risk on each eigenvalue direction of the covariance matrix for two algorithms: the original  one in Sec. C.1.1 and the coin-flip algorithm in Sec. C.2.2. The coin-flip algorithm uses two independent Poisson processes for the momentum and parameter updates in (Gen-Mom-SGD) which gives rise to much simpler ODEs.
> 3. The ODEs obtained are theoretically solvable but a lot of small high-order terms make the analysis difficult. Hence we drop high-order terms in $\lambda$ in the exact and coin-flip ODEs to obtain the ‘simplified ODEs’ (Sec. C.2.2). We prove in Sec. C.2.1 that the ‘simplified ODE’ exactly represents the evolution of the SDEs presented in Sec. 3.
> 4. Finally, using these ODEs, we can derive a Volterra equation for the full quadratic risk $\mathcal{P}(t) = \mathcal{F}(t) + (\mathcal{K} * \mathcal{P})(t)$ where $\mathcal{P}(t)$ is the risk, $\mathcal{F}(t)$ is the forcing function, $\mathcal{K}(t,s)$ is the kernel function. This is done in Sec C.1.2 and C.1.3.
>
> The rest of the proof focuses on computing asymptotics for $\mathcal{F}$ and $\mathcal{K}$ (eq. 54) by a) quantifying the empirical distribution of the covariance matrix spectrum and b) solving the ODEs.
>
> II. Quantifying the spectrum of the data covariance matrix $\hat{K}$.
> 1. We first rewrite $\mu_{F}, \mu_{K}$ in eq. 54 as integrals against two measures $\mu_\mathcal{F},\mu_\mathcal{K}$ which are defined through the deterministic equivalent in eq. 62.
> 2. Using Random Matrix Theory and previous results from [78], we provide upper and lower bounds on the deterministic equivalent measures $\mu_{F}, \mu_{K}$ in Sec. D.1 and D.2.
> 3. In Sec. D.3 and D.4 we use these bounds to define diverse components of the forcing function and kernel function (eq. 66, 68), formalized in Prop. D.13, D.14.
>
> III. We then solve the ‘simplified ODE’ eq. 53 for each class of algorithm in (Gen-Mom-SGD) separately.
> 1. For SGD-M we solve the ODE in Sec. G.1. These are the simplest as the ODE has constant coefficients, and the solutions are exponentials.
> 2. For DANA-constant, the ODE has time-varying coefficients. We employ Frobenius theory to obtain asymptotic solutions of the ODE in Sec. H.2 to H.6.
> 3. For DANA-decaying, we employ orthogonal polynomial theory to solve the ODEs in Sec. I.1.
>
> IV. Finally, we need to solve the Volterra equation for the risk.
> 1. For each algorithm to remain stable, the kernel norm has to be smaller than $1$. This leads to stability conditions on the learning rates for SGD-M (Cor. G.1), DANA-constant (lemmas H.4, H.5) and DANA-decaying (Prop. I.5).
> 2. Using this stability condition we can simplify the Volterra equation as $\mathcal{P} \asymp \mathcal{F} + \mathcal{K}$ using Kesten’s lemma (Prop. G.8, H.20, I.5).
> 3. We then compute $\mathcal{F}, \mathcal{K}$ for each algorithm by integrating the previous solutions against $\mu_{F}, \mu_{K}$ and obtain explicit scaling laws for the loss $\mathcal{P}$. This is done in Prop. G.4 for SGD-M, Prop. H.13, H.14, H.19 for DANA-constant and Prop. I.1, I.3, I.4 for DANA-decaying.
> 4. Finally, we determine the compute-optimal regime in Sec. E by a straightforward comparison of the different terms in the loss.
>
> **Alignment between data and target**
>
> We understand the reviewer’s skepticism regarding this particular assumption and asking the question whether allowing for a different structure would modify the results. First, we acknowledge that Assumption 1 and more generally the PLRF model are crude approximations of real-world scenarios. The alignment between data and target was already used in numerous previous works under the source-capacity conditions. This is the simplest model we decided to study as it is already giving rise to deep and rich behaviors as exemplified by DANA-constant and DANA-decaying stability conditions and observable in more complex systems such as LSTM where even the notion of target vector becomes ambiguous.
>
> Regarding removing this assumption towards a more general scenario of misalignment, we believe that it would indeed affect the loss curves, although it is not essential for the analysis. Indeed, the analysis only depends on the projection of the target vector in the data-covariance eigenbasis. As an example, if we multiplied the target vector by some random matrix, the expansion in the kernel eigenbasis would no longer be a pure power law as we have but saturate after some point; this could potentially result in a different scaling-law phase plane.
>
> Finally, we highlight that due to the random matrix $W$, the data covariance eigenbasis and target vector are not completely aligned. This was the main focus of the random matrix part on the deterministic equivalent, which adds the embedding bias term $\mathcal{F}_{ac}$ to the loss. This important term has a dominant role in phases 2 and 3 and was the main reason behind recovering the Chinchilla scaling in phase 3 for SGD (see [78]).
>
> **Optimality of our rates**
>
> We thank the reviewers for raising a very interesting question regarding the optimality of our rates.
>
> Within the algorithm class of Gen-mom-sgd, our theoretical analysis suggests that DANA-decaying is the stable algorithm that reaches the irreducible loss level with the least number of iterations. We kindly refer the reviewer to Section I.5 where we propose extended heuristics to derive scaling laws of all stable algorithms with the DANA class, extending results from Theorem 1. These heuristics show a continuous evolution of scaling laws from SGD to DANA-decaying and DANA-decaying to DANA-constant. This is consistent with Fig. 5 where we plotted the time to reach irreducible loss for algorithms within Gen-Mom-SGD for a specific choice of $\alpha, \beta$. We indeed observe continuous evolution of scaling laws between the three remarkable points SGD, DANA-constant and DANA-decaying, with DANA-decaying being the optimal one.
>
> The reviewer additionally raised the non-trivial question of how to generally define optimality. In [91], the authors define a class of first-order stochastic algorithms by allowing access to gradients at random data points in the same way as [73] defined for full-gradient. [91] then showed a minimax lower bound on the convergence rate by showing existence of a data distribution for which the rate for any first-order stochastic algorithm after $d$ steps is at least $\Omega(1/d)$ compared to the initialization risk, ie $\mathcal{P}(d) \gtrsim \frac{||\theta_0-\theta_*||^2}{d}$. We reported these in Table 2. As we noted in the caption of Fig. 2, “our results improve over the bounds in [91]. In particular, the two results agree as one approaches stationarity.“ Around $\alpha\approx \frac{1}{2}$, we especially recover the lower bound $\mathcal{P}(d) \gtrsim \frac{||\theta_0-\theta_*||^2}{d}$.
>
> Finally note that Gen-Mom-SGD assumes sampling a new iid data point at each iteration (as in [91] definition 4). Allowing the reuse of samples may improve sample complexity. In particular, we refer to [79] which derives improved rates by allowing multiple passes over the data. Moreover in a concurrent work [a], the authors show an improved scaling law via data reuse when the data satisfies a power-law covariance (no embedding matrix $W$) only for some choices of $\alpha, \beta$. It is a very interesting question to see if one can derive improved rates using a  DANA-decaying style algorithm with data reuse on the PLRF model. One should be able to modify the results of [77] to apply to the PLRF and compute the scaling laws similar to this work. To the best of our knowledge, this has not been done.
>
> **Limitations**
>
> We acknowledge the reviewer’s healthy skepticism regarding our assumptions and approximations and are happy that they found our empirical justification compelling. We will make it clear on lines 30-32 that we are studying a model of the problem.
>
> We tried to make every approximation clear and to provide clear definitions of the mathematical objects we use, eg ‘simplified ODEs’ (line 117 “We formulate all results going forward for the simplified ODEs“) and deterministic equivalent measures (line 132: “all the scaling law statements that follow are proven for the deterministic equivalent“). Note that the gap in the deterministic equivalent study can be removed and made entirely formal at the cost of removing the random embedding matrix $W$ which will remove the $\mathcal{F}_{ac}$ loss term. Finally, note that although approximated from the exact ODEs of Gen-Mom-SGD, the ‘simplified ODEs’ track **exactly** the evolution equation of the SDE system in Section 3, without approximation.
>
> We thank the reviewer again for their review and detailed comments that helped strengthen the paper. We hope our answer allows the reviewer to consider potentially upgrading their score if they see fit. We are also more than happy to answer any further questions.
>
> [a] Lin, Licong, Wu, Jingfeng, and Bartlett, Peter. “Improved Scaling Laws in Linear Regression via Data Reuse.”

---

> > ### Comment · Reviewer_NJzg · 2025-08-01
> >
> > I thank the authors for the clarification. I'll maintain my score (5).

---

### Official Review · Reviewer_3zYo · 2025-07-02

**Clarity:** 4
**Significance:** 3
**Originality:** 4
**Rating:** 5
**Confidence:** 3

**Summary:**

This paper offers a theoretical analysis of momentum-based stochastic optimization in high-dimensional settings, particularly when only a single pass over the data is permitted. The authors study random feature models with power-law spectra (PLRF), employing tools from random matrix theory, continuous-time dynamics, and Volterra integral equations to characterize generalization error. Their main contribution is a dimension-adaptive momentum schedule (DANA), given by \$\mu(d) = 1 - c/d^{1/3}\$, which yields provable improvements over standard SGD.

**Questions:**

1. The paper compares DANA to Adam, but why is the momentum coefficient scalar (\$\gamma\_3\$)? Would a coordinate-wise or adaptive scheme be more effective in anisotropic, high-dimensional settings?

2. The authors model gradient noise via deterministic quantities (e.g., $K\_{pp}(t)$, Appendix C, Eq. 7) and reduce the analysis to ODEs. How might an SDE-based approach, in the spirit of \[a], \[b], or \[c], alter the results or insights?

3. LSTM experiments in Section L are interesting, but serve only as indirect evidence. Could DANA be tested on architectures like CNNs or Transformers to assess its relevance in more realistic settings?

4. The paper is too long. Consider trimming the main text and moving technical appendices to a companion or journal version. The current length exceeds what is practical for peer review.

5. The Volterra formulation is elegant but dense. Would it be possible to include plots of bias terms (\$F\_{pp}\$, \$F\_{ac}\$, \$F\_0\$) and variance (\$K\_{pp}\$) over time, for varying \$\alpha\$, \$\beta\$, and \$d\$? Visualizing these could aid interpretation.

**References**

[a] Mandt, Stephan, Matthew D. Hoffman, and David M. Blei. "Stochastic gradient descent as approximate bayesian inference." Journal of Machine Learning Research 18.134 (2017): 1-35.

[b] Li, Qianxiao, and Cheng Tai. "Stochastic modified equations and adaptive stochastic gradient algorithms." International Conference on Machine Learning. PMLR, 2017.

[c] Yang, Ning, Chao Tang, and Yuhai Tu. "Stochastic gradient descent introduces an effective landscape-dependent regularization favoring flat solutions." Physical Review Letters 130.23 (2023): 237101.

**Ethical Concerns:**

["NO or VERY MINOR ethics concerns only"]

**Final Justification:**

I thank the authors for a thoughtful and comprehensive rebuttal. The authors have successfully addressed my primary concerns. The clarification in Appendix C.2.1, which links the simplified ODEs to the expected risk of an SDE model, was particularly helpful and resolved my question about the deterministic approach. Furthermore, the authors' justification for using a scalar momentum coefficient for the isotropic PLRF model is sound, and I appreciate the authors' candid discussion of this as a limitation for more general, anisotropic settings. In light of these clarifications, I have raised my score to 5.

**Limitations:**

The paper does not fully acknowledge its limitations. In particular, the assumptions of fixed features, a global momentum parameter, and deterministic dynamics constrain the analysis. The authors are encouraged to expand the limitations section and clarify to what extent DANA generalizes to modern, feature-learning models.

**Paper Formatting Concerns:**

None.

**Quality:**

3

**Strengths And Weaknesses:**

**Strengths:**

1. **Originality**: The proposed dimension-dependent momentum schedule is novel and well-justified, especially when contrasted with heuristic or fixed hyperparameter choices commonly used in SGD.

2. **Theoretical Depth**: The analysis draws on a sophisticated toolkit (Volterra integral equations, high-dimensional asymptotics, and random matrix theory) to derive interpretable bounds on generalization error.

3. **Clarity of Exposition**: Despite the technical complexity, the paper maintains a coherent high-level narrative. Key ideas are well-motivated, and derivations are presented in a readable form.

4. **Empirical Support**: Though primarily theoretical, the work includes experiments on PLRF models and linear settings that reinforce its claims.

**Weaknesses:**

1. **Excessive Length**: The paper reads more like a dissertation than a conference submission. Its length far exceeds the typical scope of a NeurIPS paper, and the sheer volume of technical material renders comprehensive verification by reviewers infeasible. While key lemmas appear sound, I did not attempt to check every derivation.

2. **Restrictive Momentum Design**: The analysis assumes a scalar momentum coefficient (\$\gamma\_3\$) applied uniformly across all dimensions. This simplification may limit DANA’s relevance in anisotropic, high-dimensional settings. The paper compares DANA to Adam, but does not explore vector-valued or coordinate-adaptive alternatives. A discussion of their feasibility could broaden DANA’s appeal.

3. **Limited Noise Modeling**: The authors model gradient noise through deterministic equivalents (e.g., \$\mathcal{K}\_{pp}(t)\$) and rely on ODE approximations. This sidesteps the stochastic dynamics that drive implicit regularization. A more direct SDE-based approach, as explored in \[a] and \[b], could better capture generalization phenomena like the preference for flatter minima \[c].

4. **Restricted Applicability**: The analysis centers on fixed-feature PLRF models, which limits insights into DANA’s performance in deep networks with learned representations. The LSTM experiments (Section L) offer some evidence, but serve mainly as a proxy. Experiments on models like CNNs or Transformers would better demonstrate practical utility.

**References**

[a] Mandt, Stephan, Matthew D. Hoffman, and David M. Blei. "Stochastic gradient descent as approximate bayesian inference." Journal of Machine Learning Research 18.134 (2017): 1-35.

[b] Li, Qianxiao, and Cheng Tai. "Stochastic modified equations and adaptive stochastic gradient algorithms." International Conference on Machine Learning. PMLR, 2017.

[c] Yang, Ning, Chao Tang, and Yuhai Tu. "Stochastic gradient descent introduces an effective landscape-dependent regularization favoring flat solutions." Physical Review Letters 130.23 (2023): 237101.

---

> ### Author Rebuttal · Authors · 2025-07-30
>
> ​​We thank the reviewer for their thoughtful comments and feedback on our work. We are especially appreciative that the reviewer views our contribution as “novel” and “well-justified”. We are also happy that the reviewer feels that our theoretical results are “sophisticated”, and that they were appreciative of the clarity “despite the technical complexity”. Building on the reviewer’s summary of our paper, we would like to take this opportunity to reiterate the core contributions of our work:
> 1. We provide **the first theoretical proof that one can change the scaling law exponent over SGD**, which requires new proof techniques in particular by including model-size dependent behaviors.
> 2. **New algorithmic development with the use of data-dependent learning rate.** Our work shows that a data-dependent learning rate, $\gamma_3(t) = (1+t)^{-1/(2\alpha)}$ can greatly improve algorithmic performance.
> 3. **Hyperparameter transferability.** The data exponent, $\alpha$ is independent of model size. Therefore, one could find the $\alpha$ on a small model and it potentially transfers to larger models without additional hyperparameter tuning (see Fig. 2, right).
> 4. **Potential applications to practical algorithms.** Many of the outscaling phenomena that we proved on the PLRF, such as $\gamma_3(t) = (1+t)^{-\kappa_3}$ living near the edge of stability persisted to LSTMs (see Fig. 2,  Fig. 18-23). We believe that this bears promise to build a competitive optimizer at scale, backed by theoretical guarantees.
>
> We now address the key clarification points raised by the reviewer:
>
> **Excessive length**
>
> We acknowledge the reviewer’s concern about the length of the manuscript. We wanted to cover many related materials to provide as much of a complete picture. In particular, we provided proofs of outscaling for many different algorithms as well as extensive experiment. These are taking a lot of space but we believe them to be core contributions of our work. We tried a lot but would be more than happy to answer any question or suggestion the reviewer may have to improve the clarity of our results. In particular, as suggested by Reviewer NJzg02, we will add a detailed proof sketch (see our answer to their review) to clarify the role of the different sections of the manuscript.
>
> **Comparison to Adam and would coordinate-wise or adaptive scheme be more effective?**
>
> We acknowledge the reviewer’s concern regarding our comparison to Adam while using a coordinate independent scalar learning rate. The features of the PLRF are isotropic, and so coordinate-wise step-sizes are not very interesting.  We mostly put the comparison (Fig. 1) because Adam is so widely used.
>
> The reviewer is correct: in an anisotropic model, a coordinate-wise optimizer would be more effective and would be an interesting subject of study.  To our knowledge there is no proven scaling law for Adam on either an anisotropic model or the PLRF.
>
> We also think a coordinate-wise optimizer, something which combines RMS-Prop and DANA, is important for practical usability of DANA and is a super direction for future research. We have some promising preliminary experimentation on transformers language models (GPT2 architectures) on the fineweb dataset that shows that mixing DANA with adaptive schemes from RMSProp shows a similar acceleration against the vanilla adaptive baseline RMSProp. We unfortunately cannot share this picture due to NeurIPS policy.
>
> **Deterministic vs SDE-based approach**
>
> The reviewer raises a very interesting question regarding the link between SDE and ODE based approaches. We agree with the reviewer’s comment that our analysis is deterministic since we study ODE on expected quantities (see sec and derivation of ODEs). We show in Appendix C.2.1 that the ‘simplified ODEs’ are the expected risk of the SDEs. There could be potentially other interesting directions of inquiry (like in [a],[b],[c]), such as how concentrated the risk curves are around the ODEs, or other properties of the distribution of the iterates under the SDE (e.g. tail behavior).
>
> **Practical relevance and test on other architectures**
>
> The reviewer raises the interesting question of the practical performance of DANA-decaying on real-world problems. We kindly highlight that our work's main contribution is a theoretical proof of the outscaling of a stochastic momentum algorithm against SGD. We additionally performed extensive experiments on the PLRF demonstrating this outscaling phenomenon, and large scale experiments on an LSTM model showing promising results of outscaling in practice with an improved scaling exponent compared to SGD. Additional promising experiments on a transformer GPT2 model suggests outscaling could occur with $\kappa_3\approx 0.7$  very similar to the one used for LSTMs. This highlights the benefit of DANA-decaying which depends on a data-dependent hyper-parameter $\alpha$ that may be easily transferable across architectures and scale (see Fig. 2 right). We suspect together with the reviewer that an adaptive version may be needed to make it competitive and believe this is an open and crucial avenue for future research.
>
> **Visualization of Volterra components throughout training**
>
> This is a great question and we can indeed simulate these. The way one would do this is that we would write down the ODEs that describe the behavior of F and K (similar to what is currently in the paper) and we can simulate these ODEs.
>
> **Limitations**
>
> We thank the reviewer for their detailed feedback and suggestions. We believe to have already addressed some of the limitations raised by the reviewer, especially on the use of **non-adaptive learning rate** (lines 262-264 “While we compare DANA against other non-adaptive momentum methods, most real-world problems benefit from adaptive methods such as Adam. Hence it would be interesting to have an analysis of DANA combined with Adam or other preconditioned methods.“) and of **fixed features, quadratic model** (line 259 “A full convergence argument beyond quadratics for DANA-decaying would be desirable”). We do not want to overclaim any of our results and will **extend the limitation section** using the reviewer’s suggestions as follows:
> * **fixed feature** Our analysis is restricted to the setting of linear regression with quadratic loss. Hence this does not incorporate feature learning which has been shown to be a critical factor of success of recent machine learning systems. Incorporating such behavior in momentum analysis, for example using multi-index models as pursued in [a] is a very interesting and non-trivial direction for future work.
> * **global momentum parameter** Our analysis uses a global learning rate. While our core contribution focuses on how to optimally scale this learning rate using model size or data dependent quantities, this has to be compared with most state-of-the-art optimizers which use per-parameter learning rate by incorporating adaptive updates. We believe that extending our analysis to adaptive learning rates is a rich and non-trivial future research direction. No work has yet derived scaling laws for adaptive SGD (Adam or RMSProp), even without momentum.
> * **deterministic dynamics** Our analysis is based on ODE description of the risk by taking expectation over the random sampling of data points. Hence our results do not capture stochastic variance of sampling particular datapoints. Extensive PLRF experiments suggest general agreement between deterministic dynamics and actual evolution but any theoretical guarantee would be a very valuable future contribution. In particular, in light of recent work [77, Theorem 1.1] we believe that at least below the high-dimensional line $2\alpha<1$, we could similarly prove concentration of the training dynamics around deterministic quantities.
>
> We thank the reviewer for their valuable feedback and great questions. We believe we have answered to the best of our ability all the great questions raised by the reviewer and we kindly ask the reviewer to potentially upgrade their score if they are satisfied with our responses. We are also more than happy to answer any further questions that arise.
>
> [a] Ren, Yunwei, et al. "Emergence and scaling laws in sgd learning of shallow neural networks." arXiv preprint arXiv:2504.19983 (2025).

---

> > ### Comment · Reviewer_3zYo · 2025-08-06
> >
> > I thank the authors for a thoughtful and comprehensive rebuttal. The authors have successfully addressed my primary concerns. The clarification in Appendix C.2.1, which links the simplified ODEs to the expected risk of an SDE model, was particularly helpful and resolved my question about the deterministic approach. Furthermore, the authors' justification for using a scalar momentum coefficient for the isotropic PLRF model is sound, and I appreciate the authors' candid discussion of this as a limitation for more general, anisotropic settings. In light of these clarifications, I have raised my score to 5.

---

### Official Review · Reviewer_P7JW · 2025-07-03

**Clarity:** 3
**Significance:** 4
**Originality:** 3
**Rating:** 5
**Confidence:** 4

**Summary:**

This paper provides a comprehensive study of the scaling laws of SGD with momentum in linear regression.

**Questions:**

1. How does the variance error occur in the problem setting? From my understanding, the variance error should stem from the label noise, but there is no label noise in this setting.
2. How does DANA-decay compare with the hyperparameter selection in Li et al. (2023) (Ref. [59])?

**Ethical Concerns:**

["NO or VERY MINOR ethics concerns only"]

**Final Justification:**

This is a paper with strong techniques, solid findings, and comprehensive discussions. After reading the authors' reply to all reviewers, I find no more confusions. The only drawback of the paper seems to be the dilemma of extensive length and limited space, but could be significantly improved given the additional page if the paper is accepted. I would strongly recommend the acceptance of this paper.

**Limitations:**

This paper does not have any outstanding limitations or negative societal impacts.

**Paper Formatting Concerns:**

The lower part of Page 5 seems a bit messy. Perhaps several figures are lacking captions.

**Quality:**

3

**Strengths And Weaknesses:**

## Strengths

1. This paper extends the scaling law in linear regression optimized with SGD to SGD with momentum acceleration.
2. The findings that SGD with heavy-ball momentum does not outscale vanilla SGD but Nesterov momentum with dimension adaptation can are impressive.
3. The authors perform extensive experiments.
4. The authors discuss the comparison of DANA against the most important variants of SGD-M extensively in the appendices.

## Weaknesses

1. The technical terms "population bias", "model capacity", "embedding bias" and "variance" are not clearly defined, although they already appear in the literature.
2. It is understandable for such an extensive study, but a little unpleasant to see that the paper has only 4 sections and no subsections at all. Section 3 is also very long, which makes the understanding of the main idea a bit more difficult.

---

> ### Author Rebuttal · Authors · 2025-07-30
>
> We would like to thank the reviewer for their time, feedback, and positive evaluation of our work which helped us strengthen the updated manuscript. We are happy to hear that the reviewer feels that our results are “impressive” and that our empirical evaluation was compelling. We now address the key clarification points raised in the review:
>
> **Clarification of technical terms**
>
> We acknowledge the reviewer’s concern regarding the definition of some technical terms which would help clarify our results. We mainly ran out of space.  We have added the following definitions in the paper:
> * *"Population bias"* ($\mathcal{F}_{pp}$): This loss term corresponds to the loss dynamics when running full-batch gradient descent on the problem (hence following the population gradient), without the embedding matrix $W$.
> * *"Model capacity"* ($\mathcal{F}_0$): This loss term (which is only $d$-dependent) represents the limit of the loss, as the number of iterations reaches infinity. It arises from the partial expressivity of our model class, since the learned parameters $\theta \in \mathbb{R}^d$ cannot encode the whole target vector $b\in \mathbb{R}^v$ when $v > d$
> * *"Embedding bias"* ($\mathcal{F}_{ac}$): This loss term comes from the random embedding matrix $W$ which deforms the spectrum of the data covariance matrix and misaligns it with the target vector $b$.
> * *"Variance"* ($\mathcal{K}_{pp}$): This loss term comes from the stochasticity of the algorithm which at each step samples a new i.i.d. random datapoint $x_t$ to compute a stochastic gradient. This stochastic gradient, whose average recovers the population gradient, is a non-exact estimate of the gradient and therefore introduces this additional loss term.
>
> **Sections unclear**
>
> We acknowledge the reviewer’s legitimate concern about the division of our sections, and more precisely Section 3. We ran out of space but will break up the sections in more precise and delimited subsections in the updated manuscript to improve clarity of the paper. In particular, we updated the manuscript to end Section 3 after the SDE part, and created a new Section 4 titled “Deriving the scaling laws of (Gen-Mom-SGD) on the PLRF model” to present our main results and Theorem 3.1.
>
> **Origin of the variance term**
>
> The reviewer raises a very interesting point regarding the origin of the variance term in the scaling laws eq.9. The reviewer is indeed correct that the variance term $\mathcal{K}_{pp}$ does not arise from label noise in our setting. The variance here comes from the stochasticity of the gradients which samples a new random point $x_t$ from the data distribution at each iteration to compute a stochastic estimate of the gradient. The gradient is a stochastic sample whose average equals the population gradient, but whose variance gives rise to an additional loss term in the scaling laws.
>
> On a different note, we kindly remark that due to model capacity limitations ($\theta \in \mathbb{R}^d$ while $b\in \mathbb{R}^v$ with $v>d$) the target label $\langle b, x\rangle$ cannot be recovered from the model $\langle W^Tx,\theta\rangle$. This creates some intrinsic label noise which in turn gives rise to the $\mathcal{F}_0$ term in the loss. This term can equivalently be seen as the model capacity limitation.
>
> Finally, we believe that introducing some additional additive noise in this model would not change the study substantially. It would mainly add a term in $\mathcal{F}_0$ and consequently would shift the loss curves. We thank the reviewer for raising this valuable question and added a remark about it in the updated manuscript.
>
> **Comparison with hyperparameter selection in Li et al. (2023)**
>
> We thank the reviewer for this interesting question. We reported extensive hyperparameters comparison with other algorithms including Li et al. (2023) in Table 3. We note that neither DANA-constant, nor DANA-decaying can be straightforwardly compared to Li et al. (2023). Indeed, as derived in the first column of Table 3, the corresponding $\Delta(t)$ in Li et al. (2023) is constant but $d$-dependent (as are other hyperparameters).  As we took $\Delta(t)$ non-$d$-dependent, we believe that the algorithm in Li et al. (2023) does not straightforwardly enter in the category of SGD-M or DANA. The case where $\Delta$ is d-dependent and learning rates d-dependent was also studied in [*] under DAHB. It was shown (not under power-law) that DAHB did not accelerate, but did improve the constant of SGD. We suspect that a similar (non-acceleration) result holds for Li et al. parameters under the power-law covariance setting.
>
> [*] C. Paquette & E. Paquette, Dynamics of Stochastic Momentum Methods on Large-scale, Quadratic Models, NeurIPS 2021
>
> **Presentation improvement page 5**
>
> We thank the reviewer for pointing this presentation issue to our attention. We originally ran out of space and reported the description of the phase space on its right. As suggested by the reviewer, we will improve the updated manuscript with clear captions to strengthen the clarity of the flow.
>
> We are grateful to the reviewer for their great questions and hope that our answers clarified them. We believe we have addressed to the best of our ability the weaknesses and questions raised by the reviewers and that they strengthened the manuscript. We politely encourage the reviewer to ask any further questions they may have and, if they are presently satisfied with our responses, to consider a fresher evaluation of our paper.

---

> > ### Comment · Reviewer_P7JW · 2025-08-02
> >
> > Many thanks for the detailed reply. I have now more confusions, and would like to remain the score of 5.

---

### Official Review · Reviewer_fBHr · 2025-07-14

**Clarity:** 3
**Significance:** 3
**Originality:** 3
**Rating:** 5
**Confidence:** 2

**Summary:**

The paper introduces DANA, a momentum algorithm that adapts its momentum and learning rate hyperparameters based on model size and data complexity. The core claim is that DANA achieves a superior scaling law exponent compared to SGD and classical SGD-M in several regimes, particularly when $2\alpha > 1$. The authors leverage a continuous-time analysis of the risk dynamics, combined with RMT, to theoretically prove these advantages. These claims are supported by extensive experiments on the synthetic PLRF model and small LSTM language models.

**Questions:**

1. Although I have not thoroughly verified the correspondence between the ODE and the discrete-time algorithm, I noticed that the SDE model (Equation (7)) is not a straightforward continuous-time counterpart of DANA, i.e., DANA does not appear to be a direct discretization of the SDE. How should I interpret their relationship? What are the errors between the continuous-time models and DANA?
2. Given that DANA shares some similarities with the algorithms listed in Table 3, I’m curious about its empirical performance on real-world tasks, particularly when hyperparameter tuning is permitted.

**Ethical Concerns:**

["NO or VERY MINOR ethics concerns only"]

**Final Justification:**

I believe the overall quality of the paper is appropriate for NeurIPS. The authors’ rebuttal addressed my questions regarding hyperparameter tuning and the outscaling regime of $2 \alpha > 0$. Therefore, I am inclined to increase my score.

**Limitations:**

yes

**Quality:**

3

**Strengths And Weaknesses:**

My evaluation of this paper is based on limited expertise in the area. Some parts of the paper may be beyond my full understanding, and my review reflects this limitation.

Strengths:
1.  The paper is well-written and clearly structured.
2.  The primary contribution, a theoretical proof that DANA can improve scaling law exponents is novel. The theoretical framework appears solid, and the claims are supported by extensive empirical validation.

Weaknesses:
1.  The theoretical analysis is limited to the PLRF model, which has a quadratic objective. While the LSTM experiments suggest it is a useful proxy, the extent to which these results generalize to the non-convex landscapes of more complex architectures remains unknown.
2.  The practical guidance is limited by the fact that the optimal DANA schedule requires knowing the exponents $\alpha$ and $\beta$, which are unknown for real-world tasks. The proposed heuristic of sweeping a new hyperparameter, $\kappa_3$, while practical, makes it less clear how DANA's tuning process compares in difficulty and outcome to other advanced optimizers that also require tuning.
3.  The theoretical proof of outscaling is established for the $2\alpha > 1$ regime. There is no direct evidence as whether this applies across modern machine learning applications.

---

> ### Author Rebuttal · Authors · 2025-07-30
>
> We thank the reviewer for their time and valuable feedback. We are heartened to hear that the reviewer finds our theoretical contribution “novel” and “solid”, and that they appreciated the “extensive empirical validation”. We thank the reviewer for considering our manuscript to be “well-written” and “clearly structured”. We now address below the key clarification points raised by the reviewer.
>
> **Limitation of the theoretical analysis to the PLRF model**
>
> We acknowledge the reviewer’s concern regarding the limitation of our analysis to the PLRF model which uses a quadratic objective. We agree that this is a clear limitation, and we put it in the “Conclusion, limitations and future work” Section lines 259-260. Going beyond this simplified model constitutes an interesting and non-trivial research question.
>
> We believe that although the PLRF model might not reflect the entire training trajectory of deep neural network’s training, different regions in the $\alpha, \beta$ plane may reflect parts of the training dynamics where for example, optimization may shift from a population dominated region (phase 1) to a more noise-dominated behavior later in training (phase 3). Hence, we hope that the PLRF may help us understand various training behaviors.
>
> Finally, we note that recent works are pointing towards very good agreement of behaviors between quadratic/convex optimization and the dynamics of complex large neural networks, see e.g. [a, b]. This encourages the study of simplified quadratic objectives as a proxy for more complex training settings.
>
> **Dependency of DANA schedules on $\alpha, \beta$ and applicability to real-world scenarios**
>
> We thank the reviewer for giving us the opportunity to strengthen the discussion regarding the dependency of our schedules to the $\alpha, \beta$ hyperparameters. We agree with the reviewer that hyperparameter tuning is a major difficulty in the training of large models and answer below in three main points:
>
> * DANA-decaying schedule **depends only on the $\alpha$** parameter, not on $\beta$. This characteristic is both surprising, and promising since $\beta$ is less tractable in real world problems. $\alpha$ on the other hand may still be measurable as the power-law decay exponent of the data covariance spectrum. Even better, DANA-constant schedule **depends neither on $\alpha$ nor on $\beta$,** but only on $d$. Note that we believe this may also have drawbacks in the training of multi-layer networks where $d$ becomes ambiguous. Characterizing the precise meaning of $d$ and $\alpha$ in the training of deep networks is a very interesting direction for future research.
> * We suspect that $\alpha$ may depend more on the data than the precise model architecture. Indeed in both LSTMs and GPT2 preliminary experiments (which we cannot share due to NeurIPS policy), the optimal $\kappa_3=1/(2\alpha)$ value is around 0.6-0.7. This bears promise for **transferability of hyperparameters**. Additionally, from our PLRF theoretical results and underscored by LSTMs results in Figure 2c, $\alpha$ does not depend on model size. We only need to measure it once and transfer to higher scales. This again needs more empirical evidence and is a very interesting direction for future research.
> * Finally even if characterizing the optimal $\alpha$ allows to have maximal acceleration on the edge of stability, note that using $\kappa_3=1/(2\alpha)$ with any $\alpha>0.5$ above the high-dimensional line will outscale SGD provided the algorithm is stable (this result for which we provide heuristics in Section I.5 was not proved but we believe it to be true using very similar proof techniques). Hence although HP tuning allows for optimal performance, DANA-decaying should bear huge advantages at scale even without optimal tuning.
>
> Finally, we agree with the reviewer that making DANA a competitive optimizer on real world problems is a highly non-trivial but very interesting direction for future research. While the main goal of our work was to provably show that outscaling is possible using a different algorithm, we suspect that to make it a competitive algorithm, DANA should be mixed with adaptive updates as in RMSProp or ADAM. For example, Schedule-Free [32] with AdamW does well on many tasks, but not Schedule-Free with SGD which is what we compared with. Preliminary experiments on GPT2 models (which we cannot share due to NeurIPS policy) show similar acceleration of DANA+RMSProp against vanilla RMSProp.
>
> **Outscaling is established for the $2\alpha>1$ regime**
>
> We acknowledge the reviewer’s healthy skepticism regarding the applicability of our outscaling results below the high-dimensional line. We mentioned it in our limitation section lines 260-261 (“It is an open question if outscaling is possible for either the 2α<1 case“) and lines 266-267 (“Defining and measuring this $\alpha$ on real-world problems, particularly determining whether $2\alpha>1$, is an important direction for future work“ but are happy to provide a more detailed discussion on it as it is a natural and interesting future research direction.
>
> First, we kindly remark to the reviewer that our analysis also shows outscaling of DANA-constant for $2\alpha>1$ for large batch $B\asymp d$. We didn’t explicitly mention this result in the main paper because of lack of space and that we believe a general theory for large batches requires more work but our analysis for DANA-constant also applies in the scaling regime $B\asymp d$ (see parametrization H.1 and remark H.1). When batch $B=1$, the reviewer is indeed right that DANA-constant only outscales SGD for $\alpha>1/2$ and reduces to SGD for $\alpha<1/2$. However, when $B\asymp d$, the stability condition (Lemma H.5) imposes $\gamma_3/(\gamma_2B)\lesssim d \iff \gamma_3 \lesssim \gamma_2$. Hence, taking $\gamma_3 \asymp \gamma_2$ and applying Theorem H.3, it becomes clear that **DANA-constant outscales SGD below the high-dimensional line** ($¼ < \alpha < ½$). We thank the reviewer for pointing out this ambiguity to us and will update the manuscript to add a remark about this. Finally, we believe that a general theoretical framework to determine the effect of large batches on all algorithms in (Gen-Mom-SGD) is a very interesting direction for future research.
>
> Experiments on LSTMs and GPT2 show a similar optimal $\kappa_3= 1/(2\alpha) \approx 0.7$ which is situated above the high-dimensional line $2\alpha>1$ and corresponds to DANA-decaying outscaling SGD. Although this requires further investigation, we believe that this exponent does not depend on the architecture but mostly on the data modality (here language). Characterizing $\alpha$ on real-world problems beyond LSTMs and GPT2, and whether acceleration is possible constitutes a very interesting research direction.
>
> **Correspondence between the ODE and the discrete-time algorithm**
>
> Interpret relationship between SDE and ODE: You should interpret the two independent Brownian motions in the SDE as using two independent gradient estimates, one for the momentum buffer $Y_t$ and one for the direct state update of $\Theta_t$. The second change is a simplification of the diffusion coefficient (i.e. the covariance matrix of the gradient update), where we effectively drop the dependence of $\langle X, \Theta_t - \Theta_*\rangle$ and the data covariance $\mathbb{E}(X \otimes X)$ – this is a correct approximation in the ‘high-dimensional case’ [references 75-77]; see also Remark C.3 where we discuss this using the ODEs.  We’re happy to expand on how this approximation relates to Gen-Mom-SGD in the paper.
>
> As for the ODEs, the ‘simplified ODEs’ exactly describe the mean risk behavior of the SDEs.  We also have a family of ‘exact ODEs’ which give the exact risk behavior of DANA when you ‘continuize it’, meaning you compute the exact risk of DANA but at a random time (see eq (17) and (18)).  See also [13]; and just to be clear, for the ‘exact ODEs’ we are using exactly the update rule of Gen-Mom-SGD – no simplifications. We judged the ‘exact ODEs’ too complicated to analyze directly, and so we created the ‘simplified ODEs’ by removing higher-order terms.
>
> **Empirical performance comparison with Algorithms in Table 3 on real datasets**
>
> It’s a great question.  We suspect that for DANA to be competitive on real problems, one needs an adaptive version (e.g., Schedule-Free with AdamW does well on many tasks, but not Schedule-Free with SGD). We have hope that such an adaptive result may be possible especially given the performance boost over SGD that we see in the LSTM experiments. An adaptive version of DANA is a future research direction we are pursuing, and we think that doing a rigorous comparison of adaptive-DANA on real-world problems is worth it.  (We have been running adaptive-DANA on GPT experiments, which unfortunately due to current NeurIPS rules, we cannot share).
>
> Moreover, almost all of the algorithms in Table 3 require knowledge of ‘model size’ and the momentum parameters have to scale with this quantity (otherwise you do not get any acceleration). Because of this dependency on model size, it might be harder to transfer knowledge of a small model’s hyperparameter to a larger model since ‘model size’ is not clearly defined for (non-PLRF models) – model size is probably not the full number of parameters on more complex architectures. We do think there’s some chance that the $\kappa_3=1/(2\alpha)$ in DANA-decaying remains stable across scales (see Figure 2).
>
> [a] Schaipp, Fabian, et al. "The surprising agreement between convex optimization theory and learning-rate scheduling for large model training." arXiv preprint arXiv:2501.18965 (2025).
>
> [b] Qiu, Shikai, et al. "Scaling Collapse Reveals Universal Dynamics in Compute-Optimally Trained Neural Networks." arXiv preprint arXiv:2507.02119 (2025).
>
> [c] Kaiyue Wen, Fantastic Optimizers and Where to Find Them (2025)

---

> > ### Comment · Reviewer_fBHr · 2025-08-04
> >
> > I appreciate the authors’ detailed rebuttal, which addresses several of my concerns and questions. In light of this, I am inclined to raise my score.

---

### Note · Authors · 2025-08-15

Dear Reviewers, AC, and SAC,

As the discussion period concludes, we wish to thank the reviewers for their valuable feedback and strong consensus in favor of our work. We are especially heartened that the reviewers find our analysis to be “impressive” (P7JW), an “important paper” (NJzg), “solid” (fBHr), and “novel” (3zYo, fBHr). We would like to take this opportunity to reiterate one last time our core contributions:
1. We provide **the first theoretical proof that one can change the scaling law exponent over SGD**. This involved using new techniques from a “sophisticated toolkit” (3zYo) to account for model-size and data–dependent effects. In contrast, we show that SGD-M fails to outscale SGD and vanilla Nesterov momentum diverges, highlighting the difficulty of changing the scaling law exponent.


2. **Data-dependent learning rate**. We propose $\gamma_3(t) = (1+t)^{-1/(2\alpha)}$, a data-dependent schedule that provably improves the loss exponent on the PLRF model.


3. **Hyperparameter transferability**. The data exponent $\alpha$ is model-size–independent, enabling tuning on small models and transfer to larger ones without additional HP-tuning (Fig. 2, right).


4. **Path to practical large-scale optimizers.** Many of the outscaling phenomena that we proved on the PLRF, such as $\gamma_3(t) = (1+t)^{-\kappa_3}$ living near the edge of stability persisted to LSTMs (see Fig. 2,  Fig. 18-23). We believe this holds promise for building a competitive optimizer at scale, backed by theoretical guarantees.
5. We perform “extensive” experiments (fBHr, P7JW, NJzg) on PLRF and LSTMs to support our claims and show a high degree of fidelity between our results and experiments. Finally, our framework allows us to **precisely conjecture the scaling behavior of many related algorithms** (Schedule-free SGD, Accelerate SGD …) which we validate experimentally.

We finally wish to reiterate our gratitude for the reviewers and AC’s work and feedback. We hope that this additional context will help clarify the paper’s significance for readers and further support the positive evaluation provided by the reviewers.

Best regards,

The authors

---

### Decision · Program_Chairs · 2025-09-17

**Decision:**

Accept (spotlight)

**Comment:**

This paper investigates the scaling law of SGD with momentum under power-law inputs. Both theoretical analysis and empirical results demonstrate that SGD with momentum yields the same scaling law exponent as plain SGD. The authors also introduce DANA, a method in which the momentum parameter adapts to the model dimension, leading to improved scaling behavior.

Most reviewers agreed that the paper deserves acceptance. After carefully reading the rebuttal and subsequent discussion, I also share this view. However, I recommend that the authors address the reviewers’ feedback in the final version of the paper, with particular emphasis on:

1-Providing more detailed explanations of the theoretical analysis.

2- Including experiments on models beyond large language models (LLMs).